# SOO-Bench: Benchmarks for Evaluating the Stability of Offline Black-Box Optimization

**Hong Qian, Yiyi Zhu, Xiang Shu, Shuo Liu, Yaolin Wen, Xin An, Huakang Lu, Aimin Zhou**[*]
East China Normal University, China
{hqian,amzhou}@cs.ecnu.edu.cn, yiyizhu@stu.ecnu.edu.cn

**Ke Tang**
Southern University of Science and Technology, China
tangk3@sustech.edu.cn

**Yang Yu**
Nanjing University, China
Polixir Technologies, China
yuy@nju.edu.cn

## Abstract

Black-box optimization aims to find the optima through building a model close to the black-box objective function based on function value evaluation. However, in many real-world tasks, such as the design of molecular formulas and mechanical structures, it is perilous, costly, or even infeasible to evaluate the objective function value of an actively sampled solution. In this situation, optimization can only be conducted via utilizing offline historical data, which yields offline black-box optimization. Different from the traditional goal that is to pursue the optimal solution, this paper emphasizes that the goal of offline optimization is to stably surpass the offline dataset during optimization procedure. Although benchmarks called Design-Bench already exist in this emerging field, it can hardly evaluate the stability of offline optimization and mainly provides real-world offline tasks and the corresponding offline datasets. To this end, this paper proposes benchmarks named SOO-Bench (i.e., Stable Offline Optimization Benchmarks) for offline black-box optimization algorithms, so as to systematically evaluate the stability of surpassing the offline dataset under different data distributions. Along with SOO-Bench, we also propose a stability indicator to measure the degree of stability. Specifically, SOO-Bench includes various real-world offline optimization tasks and offline datasets under different data distributions, involving the fields of satellites, materials science, structural mechanics, and automobile manufacturing. Empirically, baseline and state-of-the-art algorithms are tested and analyzed on SOO-Bench. Hopefully, SOO-Bench is expected to serve as a catalyst for the rapid developments of more novel and stable offline optimization methods. The code is available at https://github.com/zhuyiyi-123/SOO-Bench.

## 1 Introduction

Black-box optimization (BBO) is widely used in scientific and engineering fields, for example, hyper-parameter tuning (Snoek et al., 2012). BBO only needs function evaluation and does not require explicit function expression. To search for the optimal solution, a fundamental ingredient in model-based BBO is constructing a surrogate model to approximate the black-box objective function based on actively sampled solutions and their evaluated function values. However, in many real-world tasks, such as exploring drug molecular structures (Gaulton et al., 2012) and designing hardware mechanical structures (Yazdanbakhsh et al., 2021; Reagen et al., 2017), it is perilous, costly or even infeasible to actively sample solutions and evaluate their objective function values. In these cases, optimization can only be conducted via utilizing a limited number of offline historical data. That is to say, optimization algorithms need to make full use of the offline dataset in hand to learn the surrogate model and determine the good solution Jin et al. (2021). This yields offline model-based black-box optimization, abbreviated as offline optimization (Trabucco et al., 2021; Tan et al., 2025).

---

[*]Corresponding author.

A significant challenge in offline optimization is the narrow distribution of data within the offline dataset (Trabucco et al., 2021; Brookes et al., 2019; Fannjiang & Listgarten, 2020; Chen et al., 2022; 2023; Yu et al., 2021; Qi et al., 2022; Fu & Levine, 2021). In many real-world situations, we can only use narrow data distributions for offline optimization due to data collection strategies. For instance, when adjusting mechanical structure parameters, the offline dataset typically includes only the test data that the experimenter already possesses. However, these test data may be influenced by the subjective opinions of the experimenters and may not evenly cover the solution space.

Current research on offline optimization algorithms primarily focuses on finding the optima of a black-box function (Trabucco et al., 2021; Brookes et al., 2019; Fannjiang & Listgarten, 2020; Chen et al., 2022; 2023; Yu et al., 2021; Qi et al., 2022; Fu & Levine, 2021; Yuan et al., 2023; Zhu et al., 2025). However, the surrogate model in offline optimization can be increasingly misled in the area uncovered by the offline dataset due to the narrow distribution issue. It means that the surrogate model could greatly overestimate the objective value of optima far from the offline dataset covered regions (Trabucco et al., 2021) and results in degradation during the optimization procedure (Lu et al., 2023). Under this observation, we point out that stability is equally important as optimality for a comprehensive evaluation of offline algorithms. Herein, stability refers to the algorithm's ability to consistently surpass the offline dataset as much as possible during the optimization process without being misled by the narrow data distribution. *To the best of our knowledge, this paper is the first work to highlight stability as one of the core objectives in offline black-box optimization and conduct a systematically study of it.* Moreover, in real-world scenarios, the degree of narrow distribution varies and is difficult to be estimated beforehand. Therefore, assessing an algorithm's stability and optimality under different levels of narrow distributions is also crucial. Unfortunately, existing benchmark for offline optimization, called Design-Bench (Trabucco et al., 2022) have limitations and inabilities in these aspects. The narrow distribution in Design-Bench is artificially constructed and fixed, and there is no established indicator for evaluating the stability of offline optimization algorithms. The field of offline optimization urgently requires a benchmark capable of comprehensive algorithm evaluation in stability and optimality, so as to boost the developments of stable offline optimization (SOO) approaches.

In response to the aforementioned demands, this paper further proposes benchmarks named SOO-Bench to evaluate offline optimization algorithms' stability and optimality. Specifically, SOO-Bench provides **(1)** real-world offline optimization tasks including satellites, materials science, structural mechanics and so on, **(2)** customizable narrow distribution levels to tailor the difficulty levels of offline optimization datasets, **(3)** a novel stability indicator called Stability-Optimality indicator (SO) to measure the stability and optimality of algorithms. Besides, SOO-Bench also introduces the constrained offline optimization problems. Empirically, baseline and state-of-the-art (SOTA) algorithms are tested and analyzed on SOO-Bench. By incorporating these features, SOO-Bench is able to provide a more comprehensive evaluation of offline optimization algorithms. The codes of SOO-Bench can be found at `https://github.com/zhuyiyi-123/SOO-Bench`.

The subsequent sections respectively recap the related work, present the preliminaries, introduce the proposed SOO-Bench, depict the tasks and datasets in benchmarks, show the empirical analysis, and finally conclude the paper.

## 2 RELATED WORK

Although offline BBO has received widespread attention due to its application in real-world problems, there is currently only one comprehensive benchmark suite in this emerging field. Specifically, a benchmark platform called Design-Bench (Trabucco et al., 2022), which is unfortunately no longer maintained now, provides basic environments for offline optimization testing. It covers offline black-box optimization tasks in multiple fields, such as neural architecture search (Zoph & Le, 2017), DNA sequence design (A et al., 2016), drug discovery (Gaulton et al., 2012), and robot design (Ahn et al., 2019; Brockman et al., 2016). Although Design-Bench was launched as the first comprehensive benchmark suite, it only provides the basic interfaces for offline optimization, fails to analyze the narrow distribution of offline datasets fully, and does not provide a corresponding test environment. Besides, SDDObench (Zhong et al., 2024) evaluates streaming data-driven evolutionary algorithms, focusing on the need for standardized test suites in dynamic optimization, which distinguishes it from our approach. So, it is urgent to propose a new benchmark to address the aforementioned issues.

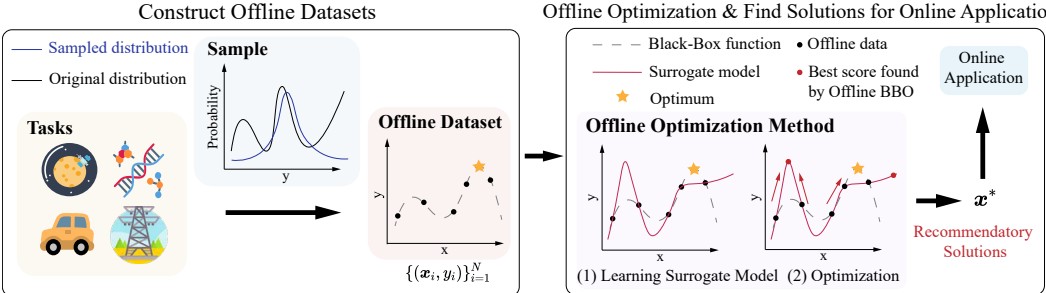

Figure 1: The process of offline black-box optimization. The left sub-figure shows the process of collecting offline data from a real black-box objective function. The right sub-figure shows the process of an offline optimization algorithm finding satisfied solutions for online application.

In order to address the narrow distribution of offline datasets, various offline black-box optimization algorithms are proposed to handle it. For example, COMs (Trabucco et al., 2021) proposes a conservative model to penalize the proxy function value far from the offline dataset solution. IOM (Qi et al., 2022) adds an inertial regularization term to force the optimized model to maintain similar representations under different data distributions, thereby alleviating the model's performance degradation on the data far from the offline dataset. CbAS (Brookes et al., 2019) models the generative model and uses a variational autoencoder (Kingma & Welling, 2014) to build a model to find the optimal solution in the trust region within an acceptable uncertainty region. CL-DDEA (Huang & Gong, 2022) introduces a contrastive learning model to enhance data-driven evolutionary algorithms by focusing on pairwise comparisons rather than absolute fitness values, while CC-DDEA (Gong et al., 2023) employs a hierarchical surrogate model and cooperative coevolution to address. Although relevant algorithms are designed to address the narrow distribution of offline datasets and attempt to find better solutions in areas near the offline datasets, there is currently no comprehensive benchmark suite available to test these algorithms comprehensively.

## 3 OFFLINE BLACK-BOX OPTIMIZATION

### 3.1 PROBLEM STATEMENT

Let $f\colon \mathcal{X} \to \mathbb{R}$ be a black-box function, where $\mathcal{X} \subseteq \mathbb{R}^d$ is a $d$-dimensional solution space. In black-box optimization, the target is to find an optimal solution $\boldsymbol{x}^*$ that maximizes $f$, which can be written as $\boldsymbol{x}^* = \arg\max_{\boldsymbol{x} \in \mathcal{X} \subseteq \mathbb{R}^d} f(\boldsymbol{x})$. $f$ is called objective function and $\mathcal{X}$ is called solution space. However, in the offline optimization scenario, direct interaction with the objective function is not allowed, and optimization can only be performed by accessing a static offline dataset $\mathcal{D} = \{\boldsymbol{x}_i, y_i\}_{i=1}^N$ containing $N$ solutions and their objective values. As shown in Figure 1, most offline optimization methods use $\mathcal{D}$ to train a model $\hat{f}_{\boldsymbol{\theta}}(\boldsymbol{x})$ to fit the objective function via supervised learning: $\boldsymbol{\theta}^* \leftarrow \arg\min_{\boldsymbol{\theta}} \sum_{i=1}^N (\hat{f}_{\boldsymbol{\theta}}(\boldsymbol{x}_i) - y_i)^2$, where $\boldsymbol{\theta}$ represents the parameters of the model. Subsequently, the solution $\boldsymbol{x}_{\mathrm{app}}$ for online optimization is found by the learned model $\hat{f}_{\boldsymbol{\theta}^*}(\boldsymbol{x})$. For example, use gradient ascent iterated $T$ times to find the optima of the surrogate model, i.e.,

$$\boldsymbol{x}^{(t+1)} \leftarrow \boldsymbol{x}^{(t)} + \eta \nabla_{\boldsymbol{x}} \hat{f}_{\boldsymbol{\theta}^*}(\boldsymbol{x})|_{\boldsymbol{x}=\boldsymbol{x}_t} , \ t = 1, 2, \ldots, T , \quad (1)$$

where $\boldsymbol{x}_{\mathrm{app}} = \boldsymbol{x}^{(T)}$ is the solution output at the terminating condition set by a certain time step $T$ for the online application. Since the exact value of $T$ at which the optimization should stop in practical applications is unknown, and in order to avoid repeatedly adjusting the number of optimization steps for different datasets, it is important that all offline black-box optimization algorithms are designed to remain stable throughout the process. This stability ensures that the worst value of the online evaluation remains relatively favorable during the optimization process.

An offline BBO method gradually learns the characteristics of the offline dataset by training a model and then inferring the online objective function. Thus, even if the offline dataset contains only local optimal solutions, the offline optimization algorithms can take advantage of the information within

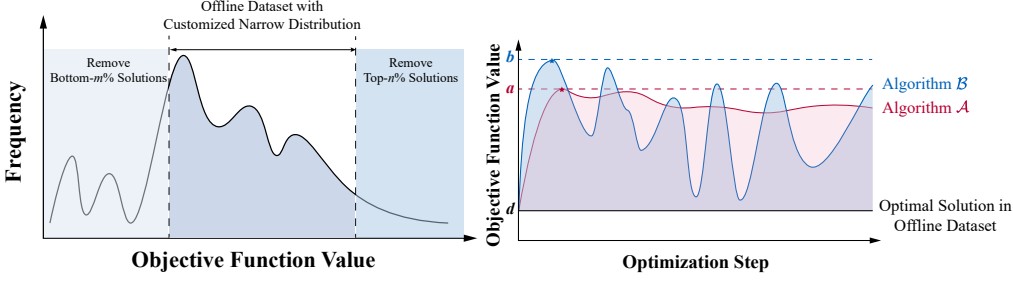

(a) An illustration of customized distribution.    (b) An illustration of SO on different algorithms.

Figure 2: The illustrations of narrow data distribution issue consider and address in SOO-Bench and the motivation of stability indicator.

the dataset, identify relations among variables, especially analyze the characteristics of these local optima, and subsequently discover new solutions that surpass the best solution in the dataset.

## 3.2 NARROW DISTRIBUTION OF OFFLINE DATASETS

In offline BBO, the data distribution of offline datasets usually does not cover the entire solution space. In real-world tasks, experimenters often obtain offline datasets through expensive real experiments. For some reasons, such as experimenters' subjectivity when constructing offline datasets—where personal biases, preferences for certain parameter settings, or prior knowledge might lead to uneven sampling—the data distribution of offline datasets cannot evenly cover all solution values in the entire solution space. For example, as shown in the "Sample" of Figure 1, the black curve represents the distribution of evaluation values of solutions uniformly sampled from the solution space, while the blue curve represents the distribution of evaluation values of the offline dataset. It can be observed that the solutions in the offline dataset mainly lie in a narrow region, which results in the out-of-distribution (OOD) issue (Kumar & Levine, 2020; Trabucco et al., 2021; Brookes et al., 2019; Chen et al., 2023) studied in previous work. As a result, some areas of the solution space lack prior data information, which causes the algorithm to be unable to model these areas accurately. This narrow distribution of offline datasets presents the greatest challenge for offline optimization problems.

Therefore, when the offline BBO methods select a solution far from the offline dataset as the optimal solution in the solution space, it must exercise extreme caution. Specifically, when providing the optimal solution, the algorithm should consider the trade-off between attempting to find a better solution to surpass the optima of the offline dataset and not moving too far away from the offline dataset to ensure stability. *Hence, the data distribution of the offline dataset directly determines the difficulty level of the offline optimization stability.* Controlling the distribution of the offline dataset is particularly significant for comprehensively testing the stability of offline BBO algorithms.

## 4 THE PROPOSED SOO-BENCH

This section introduces SOO-Bench, a more comprehensive benchmark suite to evaluate the stability of offline black-box optimization algorithms. Compared with previous work, SOO-Bench considers the customized distribution of offline datasets, allowing it to construct offline optimization tasks of different difficulties. SOO-Bench for the first time introduces the stability indicator, enabling quantitative evaluation of algorithmic stability.

### 4.1 CUSTOMIZED NARROW DISTRIBUTIONS OF OFFLINE DATASETS

The core issue of offline optimization is that the data distribution of offline datasets is usually narrow. *The narrower the data distribution covers the entire objective function search space, the more difficult the offline optimization is.* To offer a benchmark suite for comprehensive algorithmic stability and optimality performance evaluation, SOO-Bench provides offline optimization tasks with adjustable difficulty, achieved by customizing the distribution of offline datasets.

First, we provide a black-box ground-truth oracle objective function for each task. An initial dataset is obtained by uniformly sampling and evaluating the objective function. Then, the initial dataset is sorted by the value of the objective function to obtain its distribution, as shown in the curve in Figure 2 (a). Finally, datasets of different difficulty levels are constructed based on this sorted initial dataset. Specifically, the top-$n\%$ solutions by objective function value are removed to increase the difficulty of finding high-quality solutions, and the bottom-$m\%$ solutions are removed to increase the sparsity of the dataset. Through the above steps, an offline dataset with a narrow distribution in real tasks is constructed by removing solutions, as illustrated in Figure 2 (a).

## 4.2 THE PROPOSED INDICATOR FOR MEASURING STABILITY

Another problem in offline BBO is the lack of indicators to evaluate algorithm stability, which directly affects the performance of the optimal solution found by the algorithm. *We point out that the stability of an offline algorithm is reflected by its ability to continuously and stably optimize solutions that exceed the best solution in the offline dataset throughout the entire optimization process.* For example, algorithms $\mathcal{A}$ and $\mathcal{B}$ can find solutions surpassing the offline dataset's optimal solution, as shown in Figure 2 (b). However, due to the poor stability of Algorithm $\mathcal{B}$, although it found a better solution at the end of the optimization process, it failed to maintain this solution throughout the entire optimization process. In reality, when an offline BBO algorithm provides an online solution, it is often unclear which step of the optimization process should be considered. Therefore, the requirement for the algorithm to maintain a stable and high-quality solution throughout the entire optimization process becomes a critical factor affecting its effectiveness.

SOO-Bench first introduces the Stability-Optimality indicator (SO) to measure algorithmic stability. Specifically, for a task with an offline optimal solution $x^*_{\text{OFF}}$ (i.e., the best solution in the offline dataset), after $T$ optimization steps, the online evaluation value of the optimal solution provided by the algorithm is denoted as $f(x_t)$, where $t = 1, 2, \ldots, T$, and SO is then defined as follows:

$$\text{SO} = \frac{2 \cdot OI(t) \cdot SI(t)}{SI(t) + OI(t)}, \tag{2}$$

$$\text{and } OI(t) = \frac{S}{S_1}, \ SI(t) = \frac{S}{S_2}, \tag{3}$$

where $S = \sum_{t=1}^{T} f(x_i)$ represents the cumulative sum of the evaluation values curve, $S_1 = T \cdot f(x^*_{\text{OFF}})$ represents the product of the evaluation value of the optimal offline solution and total optimization steps $T$, and $S_2 = T \cdot \max_t f(x_t)$ represents the product of the evaluation value of the optimal solution found by the algorithm and total optimization steps $T$.

Intuitively, as shown in Figure 2 (b), the Optimality Indicator (OI) reflects the ratio of the area under the algorithm's evaluation curve to the area under the offline optimal solution curve, indicating the algorithm's optimality. On the other hand, the Stability Indicator (SI) represents the ratio of the area under the algorithm's evaluation curve to the area under the best solution found by the algorithm, reflecting stability by measuring how closely the algorithm's performance aligns with the ideal solution. When evaluation values are stable and approach the optimal value, the SI value nears 1, signaling better stability. Conversely, significant fluctuations or deviations from the optimal value result in a lower SI. The OI value increases as the algorithm approaches the global optimum and decreases when the algorithm's performance is consistently below the optimal offline solution. Since stability refers to the algorithm's ability to consistently surpass the offline dataset as much as possible during the optimization process, we must take both OI and SI into consideration. This results in the proposed Stability-Optimality indicator (SO) to measure algorithmic stability, as shown in Eq. (2).

Sometimes, when we assign different weights to OI and SI, indicating our degree of preference of OI and SI, it is expressed as

$$\text{SO}_\omega = \frac{OI(t) \cdot SI(t)}{\omega(t) \cdot SI(t) + (1 - \omega(t)) \cdot OI(t)}, \tag{4}$$

where $\omega \in (0, 1)$. If $\omega = 0.5$, $\text{SO}_\omega$ degenerates to SO. If $\omega > 0.5$, the significance of OI is greater. If $\omega < 0.5$, the significance of SI is greater. When $\omega > 0.5$, to improve $\text{SO}_\omega$, a desirable offline optimization algorithm should prioritize the improvement of OI. This means that the offline optimization algorithm should place more emphasis on the algorithm's ability of optimality. When

Table 1: An overview of the tasks in the proposed SOO-Bench. SOO-Bench includes several tasks, which are divided into four benchmarks. These tasks have discrete and continuous design spaces and constrained and unconstrained situations to suit offline black-box optimization. $N_{\text{var}}$ indicates the number of variables and $N_{\text{con}}$ means the number of constraints. The symbol "/" means that the global optimal function value of the real-world black-box function is unknown.

| Benchmarks | Tasks | | $N_{\text{var}}$ | $N_{\text{con}}$ | Optimum |
|---|---|---|---|---|---|
| GTOPX | gtopx 1 | Cassini 1 | 6 | 4 | 4.9307 |
| | gtopx 2 | Cassini 2 | 22 | 0 | 8.383 |
| | gtopx 3 | Messenger (reduced) | 18 | 0 | 8.6299 |
| | gtopx 4 | Messenger (full) | 26 | 0 | 1.9579 |
| | gtopx 5 | GTOC 1 | 8 | 6 | -1581950 |
| | gtopx 6 | Rosetta | 22 | 0 | -1.3433 |
| | gtopx 7 | Cassini 1-MINLP | 10 | 4 | 3.5007 |
| CEC | cec 1 | Optimal operation | 7 | 14 | -4.52912 |
| | cec 2 | Process flow | 3 | 3 | 1.07654 |
| | cec 3 | Process synthesis | 2 | 2 | 2 |
| | cec 4 | Three-bar | 2 | 3 | 2.63896 |
| | cec 5 | Welded beam | 4 | 5 | 1.67022 |
| PROTEIN | protein 1 | TF Bind 8 | 8 | 0 | / |
| | protein 2 | TF Bind 10 | 10 | 0 | / |

$\omega < 0.5$, a desirable offline optimization algorithm should place more emphasis on the algorithm's ability of stability. In this work, we employ a linearly decreasing weight function, i.e., $\omega(t) = \max(0, 1 - \frac{t}{T+1})$. This setting of $\omega(t)$ ensures that a desirable offline black-box optimization algorithm should focus on improving optimality (i.e., OI) during the early steps of optimization process, while in the later steps it should shift its priority to stability (i.e., SI). This procedure can prevent the performance degradation caused by out-of-distribution issue. The setting of $\omega(t)$ can be adjusted according to the user's specific needs.

### 4.3 Extensive Tasks, Datasets and Baselines for Evaluating Stability

SOO-Bench provides an extensive real-world benchmark environment, and all task details are introduced in Section 5. Besides, SOO-Bench reproduces a variety of baseline algorithms and tests their performance through experiments, which are introduced in Section 6. We provide users with more customizable APIs. In addition to controlling task difficulty by adjusting the distribution of offline datasets, users can also create noisy offline datasets to test algorithm performance specifically.

## 5 Tasks and Datasets Description

This section presents the details of all tasks and datasets. An overview is shown in Table 1. Despite the tasks vary a lot, we provide a unified API on our datasets. Each task in SOO-Bench provides a ground-truth oracle objective function $f(\boldsymbol{x})$. In addition to offering basic offline datasets, it also includes an API for constructing offline datasets, allowing users to create datasets with different difficulties. We provide a detailed description of the tasks in the four benchmarks in Appendix A, which is placed in the supplementary material. Our benchmarks and example implementations are available at https://github.com/zhuyiyi-123/SOO-Bench.

**GTOPX: Space Mission Optimization** (Schlueter et al., 2021). This paper addresses a set of real-world space mission trajectory problems (European Space Agency and Advanced Concepts Team, 2020), designed as numerical black-box optimization problems. The objective is to minimize the total velocity variation during interplanetary space missions. Our benchmark consists of seven GTOPX tasks. Satellite missions are particularly challenging, especially when the oracle evaluation encounters an invalid value, which returns as NaN, we indicate it as the worst value in the offline dataset. If the solution found by the offline optimization algorithm results in such a value, the true

objective function value is poor, and the algorithm proves to be very unstable. The license for this dataset is GNU General Public License.

**CEC Task: Industrial and Design Optimization** (Kumar et al., 2020). We select five real-world constrained problems, which are maximization tasks with constraints that are feasible when they are greater than or equal to zero. Constrained tasks are crucial in real life. When normalizing or denormalizing these tasks, the solutions found may exceed the variable boundaries. If some methods are applied to set bounds, the solutions may be suboptimal. Even if the algorithm demonstrates good stability, if the optimization results are poor, the overall outcome is still considered unsatisfactory. We aim for the offline optimization algorithm to find high-quality solutions while maintaining stability throughout the optimization process. The license for this dataset is CC-BY 4.0 License.

**PROTEIN: DNA Sequence Optimization** (Trabucco et al., 2022). The variable space comprises sequences of four categorical variables. The objective of the two tasks is to identify the optimal 8-nucleotide DNA sequence that exhibits the highest binding affinity to a specific transcription factor. Discrete tasks play a crucial role in offline optimization. However, in these tasks, offline optimization may identify non-existent molecules. The license for this dataset is MIT License.

## 6 EXPERIMENTS

To simulate a more realistic data distribution, we choose a dataset size that is 1000 times the variable dimension. At the same time, to further simulate the narrow distribution, the missing $m\%$ near the worst objective function value and the missing $n\%$ near the optimal objective function value are used to ensure that the data volume is small and missing near the optimal value, as shown in Figure 2 (a). In this paper, we select the middle 50% of the data (i.e., $m\% - n\% = 50\%$) to construct a simulated dataset as a reasonable baseline without leveraging any prior knowledge. Furthermore, when constructing the constraint dataset, we set the ratio of satisfying constraints to not satisfying constraints as $2 : 3$. Given the high flexibility and customizability of the proposed benchmark, users can modify the data volume, $m\%$, $n\%$, and constraint proportions as needed. Meanwhile, the proposed benchmark can adjust both the data volume size and the distribution of solutions in the offline dataset. For more details, please refer to our Github.

### 6.1 OFFLINE OPTIMIZATION ALGORITHMS

We test a range of baselines and SOTA offline optimization algorithms on each of task. Specifically, we compare with two categories of algorithms: (1) those that address unconstrained problems, including classic baselines: BO-qEI (Wilson et al., 2017), CMA-ES (Hansen, 2006), and Offline BBO methods: autofocusing CbAS (Fannjiang & Listgarten, 2020), TTDDEA (Huang et al., 2021), ARCOO (Lu et al., 2023), Tri-mentoring (Chen et al., 2023), and (2) those that address constrained problems, including CARCOO, CCOMs, DDEA-PF, DDEA-SPF (Huang & Wang, 2021a). Since classical methods lack specific methodologies for offline optimization problems, they cannot be directly applied to such problems. Therefore, we introduce a guided training agent model to provide optimization guidance through agent prediction, thereby indirectly solving the offline optimization problem. In this section, we briefly discuss these algorithms and evaluate them in the next section. Due to page limitation, the implementation codes links are listed in Appendix B.

**BO-qEI:** Offline Bayesian optimization is performed to minimize a learned surrogate function $\hat{f}(\boldsymbol{x})$ by training an ensemble of neural network models. Candidate solutions are generated with a Gaussian Process model and labeled with values from $\hat{f}(\boldsymbol{x})$. We employ the quasi-expected Improvement (qEI) acquisition function within the BoTorch framework (Balandat et al., 2019). **CMA-ES:** We compute the value of learned surrogate function $\hat{f}(\boldsymbol{x})$ on the samples $\boldsymbol{x}_t$, which is obtained from the distribution $\mathcal{N}(\mu_t, \Sigma_t)$ at an iteration $t$. We then adapt the covariance matrix to refine the belief distribution, repeating this process multiple times. **Autofocusing CbAS:** Autofocusing CbAS learns a density model $p_0(\boldsymbol{x})$ of $\boldsymbol{x}$ to approximate the data distribution, then gradually use importance sampling to re-training $\hat{f}(\boldsymbol{x})$ under current distribution $p_t(\boldsymbol{x})$, and adapts it towards the optimized solution. **ARCOO:** ARCOO constructs the surrogate model $\hat{f}(\boldsymbol{x})$ combined with the energy model, which is used to characterize the risk of degradation. After construction, a risk suppression factor is applied to control the risk. **Tri-mentoring:** This approach constructs three

Table 2: Overall results for GTOPX unconstrained tasks with 128 solutions and 100th percentile evaluations are averaged over eight runs. $f(\boldsymbol{x}^*_{\text{OFF}})$ represents the optimal objective function value in the offline dataset. FS (i.e., final score) refers to the function value found by an offline optimization algorithm at $t = 50$ during the optimization process, which consists of a total of $T = 150$ optimization steps. In this case, 25% of the values are missing near the worst value and another 25% near the optimal value.

| $f(\boldsymbol{x}^*_{\text{OFF}})$ | GTOPX 2 $-195.44_{\pm1.38}$ | | | | | GTOPX 3 $-151.85_{\pm0.06}$ | | | | | GTOPX 4 $-215.74_{\pm1.18}$ | | | | | GTOPX 6 $-112.11_{\pm0.33}$ | | | | |
|---|---|---|---|---|---|---|---|---|---|---|---|---|---|---|---|---|---|---|---|---|
| Metric | FS | SI | OI | SO | $SO_\omega$ | FS | SI | OI | SO | $SO_\omega$ | FS | SI | OI | SO | $SO_\omega$ | FS | SI | OI | SO | $SO_\omega$ |
| ARCOO | $-68.68_{\pm6.13}$ | $0.92_{\pm0.01}$ | $1.76_{\pm0.08}$ | $1.21_{\pm0.02}$ | $1.35_{\pm0.02}$ | $-47.38_{\pm7.84}$ | $0.94_{\pm0.02}$ | $1.84_{\pm0.05}$ | $1.25_{\pm0.02}$ | $1.40_{\pm0.01}$ | $-87.65_{\pm14.50}$ | $0.93_{\pm0.03}$ | $1.85_{\pm0.06}$ | $1.24_{\pm0.04}$ | $1.40_{\pm0.04}$ | $-49.43_{\pm10.25}$ | $0.91_{\pm0.03}$ | $1.79_{\pm0.07}$ | $1.21_{\pm0.03}$ | $1.36_{\pm0.06}$ |
| BO | $-100.67_{\pm17.26}$ | $0.86_{\pm0.02}$ | $1.66_{\pm0.02}$ | $1.13_{\pm0.02}$ | $1.27_{\pm0.02}$ | $-57.83_{\pm12.15}$ | $0.86_{\pm0.01}$ | $1.70_{\pm0.03}$ | $1.14_{\pm0.01}$ | $1.29_{\pm0.02}$ | $-115.19_{\pm15.73}$ | $0.85_{\pm0.02}$ | $1.65_{\pm0.04}$ | $1.13_{\pm0.02}$ | $1.26_{\pm0.00}$ | $-61.97_{\pm9.22}$ | $0.84_{\pm0.01}$ | $1.63_{\pm0.03}$ | $1.11_{\pm0.02}$ | $1.24_{\pm0.02}$ |
| CBAS | $-195.44_{\pm1.38}$ | $1.00_{\pm0.00}$ | $1.00_{\pm0.00}$ | $1.00_{\pm0.00}$ | $1.00_{\pm0.00}$ | $-151.85_{\pm0.06}$ | $1.00_{\pm0.00}$ | $1.00_{\pm0.00}$ | $1.00_{\pm0.00}$ | $1.00_{\pm0.00}$ | $-215.74_{\pm1.18}$ | $1.00_{\pm0.00}$ | $1.00_{\pm0.00}$ | $1.00_{\pm0.00}$ | $1.00_{\pm0.00}$ | $-112.11_{\pm0.33}$ | $1.00_{\pm0.00}$ | $1.00_{\pm0.00}$ | $1.00_{\pm0.00}$ | $1.00_{\pm0.00}$ |
| CCDDEA | $-166.15_{\pm41.96}$ | $0.63_{\pm0.15}$ | $1.23_{\pm0.30}$ | $0.83_{\pm0.28}$ | $0.93_{\pm0.25}$ | $-5.93e4_{\pm3.56e5}$ | $-5.04_{\pm8.09}$ | $-10.09_{\pm16.17}$ | $-6.72_{\pm10.78}$ | $-7.57_{\pm12.15}$ | $-379.37_{\pm357.55}$ | $-5.48_{\pm11.42}$ | $-10.96_{\pm22.61}$ | $-7.31_{\pm15.22}$ | $-8.24_{\pm17.15}$ | $-150.90_{\pm99.16}$ | $0.27_{\pm0.21}$ | $0.53_{\pm0.42}$ | $0.35_{\pm0.28}$ | $0.40_{\pm0.32}$ |
| CMAES | $-57.06_{\pm9.42}$ | $0.96_{\pm0.02}$ | $1.93_{\pm0.04}$ | $1.29_{\pm0.03}$ | $1.45_{\pm0.03}$ | $-39.41_{\pm4.45}$ | $0.97_{\pm0.01}$ | $1.94_{\pm0.02}$ | $1.29_{\pm0.01}$ | $1.45_{\pm0.02}$ | $-95.48_{\pm19.95}$ | $0.91_{\pm0.03}$ | $1.83_{\pm0.06}$ | $1.22_{\pm0.04}$ | $1.37_{\pm0.04}$ | $-46.42_{\pm8.38}$ | $0.92_{\pm0.03}$ | $1.84_{\pm0.06}$ | $1.23_{\pm0.04}$ | $1.38_{\pm0.04}$ |
| TTDDEA | $-245.65_{\pm47.69}$ | $0.45_{\pm0.11}$ | $0.91_{\pm0.22}$ | $0.60_{\pm0.15}$ | $0.68_{\pm0.17}$ | $-157.10_{\pm42.75}$ | $0.67_{\pm0.23}$ | $1.34_{\pm0.42}$ | $0.90_{\pm0.28}$ | $1.01_{\pm0.32}$ | $-259.66_{\pm132.15}$ | $0.43_{\pm0.44}$ | $0.87_{\pm0.88}$ | $0.58_{\pm0.59}$ | $0.65_{\pm0.66}$ | $-132.26_{\pm51.65}$ | $0.44_{\pm0.29}$ | $0.88_{\pm0.58}$ | $0.59_{\pm0.39}$ | $0.66_{\pm0.44}$ |
| Trimentoring | $-75.48_{\pm45.69}$ | $0.98_{\pm0.01}$ | $1.83_{\pm0.31}$ | $1.26_{\pm0.19}$ | $1.40_{\pm0.15}$ | $-109.04_{\pm30.67}$ | $0.99_{\pm0.02}$ | $1.48_{\pm0.47}$ | $1.15_{\pm0.15}$ | $1.23_{\pm0.23}$ | $-139.48_{\pm57.84}$ | $0.94_{\pm0.05}$ | $1.63_{\pm0.38}$ | $1.17_{\pm0.12}$ | $1.29_{\pm0.14}$ | $-82.74_{\pm30.03}$ | $0.85_{\pm0.18}$ | $1.45_{\pm0.41}$ | $1.05_{\pm0.22}$ | $1.15_{\pm0.26}$ |

Table 3: Overall results for GTOPX constrained tasks with 128 solutions and 100th percentile evaluations. In this case, 25% of the values are missing near the worst value and another 25% near the optimal value. Details are the same as Table 2.

| $f(\boldsymbol{x}^*_{\text{OFF}})$ | GTOPX 1 $-75.14_{\pm0.98}$ | | | | | GTOPX 5 $3.63e4_{\pm1.25e3}$ | | | | | GTOPX 7 $-346.52_{\pm4.33}$ | | | | |
|---|---|---|---|---|---|---|---|---|---|---|---|---|---|---|---|
| Metric | FS | SI | OI | SO | $SO_\omega$ | FS | SI | OI | SO | $SO_\omega$ | FS | SI | OI | SO | $SO_\omega$ |
| CARCOO | $-75.10_{\pm1.00}$ | $0.94_{\pm0.15}$ | $0.97_{\pm0.14}$ | $0.95_{\pm0.14}$ | $0.96_{\pm0.13}$ | $1.11e5_{\pm4.87e4}$ | $0.78_{\pm0.06}$ | $1.35_{\pm0.10}$ | $0.99_{\pm0.02}$ | $1.09_{\pm0.03}$ | $-649.96_{\pm303.32}$ | $0.58_{\pm0.33}$ | $0.91_{\pm0.45}$ | $0.69_{\pm0.35}$ | $0.74_{\pm0.37}$ |
| DEPF | $-201.33_{\pm108.62}$ | $-2.74e3_{\pm7.22e3}$ | $-2.76e3_{\pm7.21e3}$ | $-2.75e3_{\pm7.21e3}$ | $-2.75e3_{\pm7.21e3}$ | $0.00_{\pm0.00}$ | $-3.59e6_{\pm1.23e5}$ | $-3.59e6_{\pm1.23e5}$ | $-3.59e6_{\pm1.23e5}$ | $-3.59e6_{\pm1.23e5}$ | $-836.74_{\pm310.44}$ | $-2.97e4_{\pm2.97e4}$ | $-2.97e4_{\pm2.97e4}$ | $-2.97e4_{\pm2.97e4}$ | $-2.97e4_{\pm2.97e4}$ |
| DESPF | $-283.39_{\pm14.47}$ | $-1.00e4_{\pm1.00e4}$ | $-1.00e4_{\pm1.00e4}$ | $-1.00e4_{\pm1.00e4}$ | $-1.00e4_{\pm1.00e4}$ | $0.00_{\pm0.00}$ | $-3.59e6_{\pm1.23e5}$ | $-3.59e6_{\pm1.23e5}$ | $-3.59e6_{\pm1.23e5}$ | $-3.59e6_{\pm1.23e5}$ | $-954.35_{\pm3.16}$ | $-4.42e4_{\pm2.56e4}$ | $-4.42e4_{\pm2.56e4}$ | $-4.42e4_{\pm2.56e4}$ | $-4.42e4_{\pm2.56e4}$ |
| PRIME | $-260.57_{\pm44.67}$ | $-0.52_{\pm0.47}$ | $-1.03_{\pm0.94}$ | $-0.69_{\pm0.63}$ | $-0.78_{\pm0.71}$ | $0.00_{\pm0.00}$ | $0.11_{\pm0.34}$ | $0.21_{\pm0.67}$ | $0.14_{\pm0.45}$ | $0.16_{\pm0.51}$ | $-954.35_{\pm3.16}$ | $-0.38_{\pm0.43}$ | $-0.75_{\pm0.88}$ | $-0.50_{\pm0.58}$ | $-0.57_{\pm0.66}$ |

surrogate models and uses ranking supervision signals for mutual mentoring. After that, we can use adaptive soft-labeling to learn more accurate labels. **TTDDEA:** By dividing the offline dataset into three equal parts randomly, we build three surrogate models respectively. We select a high-confidence pseudo-label to fill in each other's datasets, then retrain the surrogate model, generate offspring, and repeat this multiple times. **CCDDEA:** CCDDEA designs a hierarchical surrogate-joint learning model to be able to guide the evolving population to search at different granularities and then optimize at the global and local subspace levels in a cooperatively coordinated evolutionary manner. **CARCOO:** This is a simple constrained version of ARCOO. We incorporate the degree of constraint violation into the risk assessment and consider solutions that violate the constraints as high risk. **CCOMs:** This is a reproduction version of the PRIME (Kumar et al., 2022) method. Since we do not find its codes, we simply implement PRIME according to the paper, that is, making the objective function value that violates the constraint as bad as possible. **DDEA-PF & DDEA-SPF:** It combines common constraint processing techniques with offline data-driven evolutionary algorithms to handle constrained optimization problems. This is achieved by constructing proxy models for constraints and objective functions, respectively, and using constraint processing techniques to process constraint proxy models and objective proxy models.

## 6.2 RESULTS AND ANALYSIS

This scenario represents normal settings in real-world conditions where the maximum and minimum values of the objective values are proportionally missing. We conduct a total of $T = 150$ optimization steps. In the main experiment, we present results only at optimization steps $t = 50$, with additional results available in the Appendix C. In the unconstrained optimization case, we have four tasks in GTOPX. As shown in Table 2, due to space limitations, we only present the table for $n = 25\%$ and $m = 75\%$, from which two key observations can be obtained.

● Baseline methods like BO-qEI and CMAES have shown the ability to compete with offline BBO methods in terms of optimality and stability. This result is consistent with the results in the Design Bench Trabucco et al. (2022).

● Existing offline BBO methods (e.g., ARCOO and Tri-mentoring) perform well across different benchmarks and different distributions, such as gtopx 2 and gtopx 4 in table 2.

● TTDDEA and CCDDEA can perform well on different tasks, except for gtopx 2 and gtopx 3, and even CCDDEA is the best algorithm on mujoco 2 task in Appendix C, which is because the space of gtopx 2 and 3 is more complicated.

In the constrained optimization case, we have three tasks in GTOPX. As shown in Table 3, we can obtain two key observations:

• Experimental results show that simply constructed offline BBO methods for handling constraint problems are successful on both unconstrained and constrained tasks. However, we find that DE-PF and DE-SPF perform poorly on satellite missions. Poor SO and OI values indicate that the algorithm is prone to fall into infeasible solutions during the optimization process. This result suggests that directly combining online constraint handling techniques with offline evolutionary methods may have difficulty capturing features in complex tasks with sufficient accuracy.

• We find that simply modifying the previous unconstrained offline optimization method cannot achieve good optimization results and stability on complex satellite missions, which shows that a better optimization algorithm needs to be proposed.

In real-world applications, we not only pursue the optimal solution (such as the 100th percentile), but may also choose the median (50th) or a worse solution (0th percentile), especially when facing uncertainty or limited resources. At this time, the stability of the optimization algorithm is particularly important. Even if a worse solution is chosen, we hope that the algorithm can stably improve the results and avoid the instability or risks caused by over-optimization. For example, in the drug design task, we hope that even the worst solution is relatively good to avoid the drug from having a bad effect on the human body.

In real-world scenarios, we often do not know the distribution of the dataset. We simulate this by keeping the total missing proportion constant while varying n% and m%. Based on Appendix C, the optimization stability of BO-qEI gradually decreases as n% increases. In contrast, ARCOO initially even shows some improvement but generally remains stable. Tri-mentoring performs best at $n = 20\%, m = 30\%$, but its performance is subpar at other times. In unconstrained benchmarks, we can obtain three observations:

• Classical baselines like BO-qEI, which originate from online optimization scenarios, are highly sensitive to data missing in the optimal search region. In contrast, offline BBO methods are relatively stable, demonstrating their advantage in different narrow distributions and validating their effectiveness in real-world applications.

• After 150 steps of optimization, ARCOO often achieves the best performance despite experiencing some degradation. This can be attributed to ARCOO's use of a learned energy model to assess the risk of generating solutions during the optimization process. By leveraging this risk to control the step size during gradient ascent, ARCOO maintains a certain degree of stable performance improvement.

• we observed that while some algorithms exhibited strong stability, their performance tended to degrade as the number of optimization steps increased (i.e., from 100 steps to 150 steps). This decline can be attributed to these algorithms encountering OOD regions as the optimization progresses, leading to poorer results.

• Although CCOMs are relatively stable in some cases, they often find poor solutions and stagnate, indicated by sub-optimal optimization results. The reason may be that the constraints are too strict, making the algorithm less inclined to explore outside the current solution space.

## 7 CONCLUSION AND DISCUSSION

This paper proposes SOO-Bench, a benchmark suite to promote stable offline black-box optimization. SOO-Bench provides a variety of real offline optimization tasks. Considering that the narrow distribution of offline datasets is a vital challenge of offline optimization, SOO-Bench provides APIs that can customize the distribution of offline datasets to construct tasks with different difficulties, thereby comprehensively testing the performance of algorithms. Notably, SOO-Bench emphasizes the significance of stability of offline BBO and introduces an indicator called Stability Improvement (SI) to quantify the stability. Finally, SOO-Bench reproduces offline optimization baselines and conducts experiments to evaluate algorithmic stability and optimality.

There are still some limitations of SOO-Bench. First, the variety of tasks we provide is still not rich enough, and there are many unexplored black-box optimization scenarios (e.g., robotic control). Second, we emphasize that optimization stability is crucial in the offline optimization, and the proposed indicator can trade off between the optimality of the output solution and the stability of the optimization process. But we believe that other reasonable indicators still exist. The future work will explore real-world scenarios by providing new datasets and tasks and investigate a range of data

size percentages (i.e., different degrees about the missing $m\%$ near the worst value and the missing $n\%$ near the optimal value) to conduct a more comprehensive analysis. Meanwhile, we will further improve SOO-Bench, develop more realistic narrow distribution methods to simulate real offline datasets, and propose a comprehensive evaluation system containing richer evaluation indicators.

## ACKNOWLEDGMENTS

The authors would like to thank the anonymous reviewers for their helpful and valuable comments. This work is supported by the National Natural Science Foundation of China (No. 62476091).

## ETHICS AND REPRODUCIBILITY STATEMENTS

**Ethics.** This paper does not involve human subjects, personal data, or sensitive information. All datasets used for testing are publicly available, and no proprietary or confidential information has been utilized. We take responsibility for any potential violation of rights and for ensuring compliance with data licensing requirements.

**Reproducibility.** Experimental settings are described in Section 6.1 with further details of the methods included in Appendix A-D. The datasets utilized in this paper are all publicly available and open-source. The link to our anonymized repository that includes codes, datasets, documents, demo and license can be found at https://github.com/zhuyiyi-123/SOO-Bench.

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

APPENDIX

This appendix is structured into four sections to improve understanding and facilitate the reproduction of our benchmark research. Task Description Section elaborates on the objective function, constraints, and variables used in our study. The Experimental Details Section provides comprehensive information on the methodologies, code implementations, and resource usage of all methods compared. Experimental Results Section thoroughly discusses the research results and findings in detail. Finally, the Hyperparameter Analysis Section itemizes the specific settings and configurations utilized in the experiments. Our benchmark and comparison algorithms are available at `https://github.com/zhuyiyi-123/SOO-Bench`.

## A  TASK DESCRIPTION

In this section, we detail task description in SOO-Bench. Specifically, we answer **(1)** what is the goal and background of each task? **(2)** where are the tasks from? **(3)** what do the variables and constraints in each task represent?

### A.1  GTOPX: SPACE MISSION BENCHMARKS

For those interested in the task, more information can be found on the official website for the benchmark. To access detailed descriptions, methodologies, and data related to the task, please visit the following URL: `http://www.midaco-solver.com/index.php/about/benchmarks/gtopx`. This page provides essential resources for understanding the scope and requirements of the benchmark challenges.

#### A.1.1  GTOPX 1: CASSINI 1

The Cassini 1 benchmark simulates an interplanetary mission targeting Saturn, with the mission's aim being to enter a specific orbit around Saturn characterized by an intracentric radius of 108,950 km and an eccentricity of 0.98. The primary goal of this benchmark is to minimize the total $\Delta V$, or change in velocity, required throughout the mission, which includes the launch and orbital insertion maneuvers. This scenario utilizes six decision variables and incorporates four constraints that set maximum limits on the proximity of the center during four flyby maneuvers. The variables' descriptions of this task include $var_1$ and $var_2$.

#### A.1.2  GTOPX 2: CASSINI 2

The Cassini 2 benchmark, which models complex interplanetary missions to Saturn, including critical maneuvers like the Deep Space Maneuver (DSM), presents a more challenging scenario than the gtopx 1 benchmark. Unlike Cassini 1, which focuses on orbital insertion, the primary objective of this benchmark is to achieve a rendezvous with Saturn, aiming to minimize the total $\Delta V$ required throughout the mission. This benchmark involves handling 22 decision variables. The variables' descriptions of this task include $var_1$, $var_2$, $var_3$, $var_4$, $var_5$, $var_6$, and $var_7$.

#### A.1.3  GTOPX 3: MESSENGER (REDUCED)

The third benchmark, known as Messenger (reduced), is a simulation of interplanetary missions to Mercury, excluding any resonant flybys of the planet. The main objective of this benchmark is to minimize the total $\Delta V$ over the course of the mission. It involves 18 decision variables and the explanation of the variables is given in Table 4.

#### A.1.4  GTOPX 4: MESSENGER (FULL)

The fourth benchmark, titled Messenger (full), models interplanetary missions to Mercury, incorporating resonant flybys of the planet. This benchmark aims to minimize the total $\Delta V$ incurred throughout the mission. It features 26 decision variables, which includes all variables in Table 4.

Table 4: Description of variables in GTOPX tasks (Schlueter et al., 2021).

| Variable | Descriptions | gtopx 1 | gtopx 2 |
|---|---|---|---|
| $var_1$ | Initial day measured from 1-Jan 2000 | ✓ | ✓ |
| $var_2$ | Time interval between events (e.g. departure, fly-by, capture) | ✓ | ✓ |
| $var_3$ | Initial excess hyperbolic speed (km/S) | ✗ | ✓ |
| $var_4$ | Angles of excess velocity in a hyperbolic trajectory | ✗ | ✓ |
| $var_5$ | Fraction of the time interval after which DSM occurs | ✗ | ✓ |
| $var_6$ | Radius of flyby (in planet radii) | ✗ | ✓ |
| $var_7$ | Orientation of the trajectory angle in the planet's B-plane approach vector | ✗ | ✓ |

### A.1.5  GTOPX 5: GTOC1

The fifth benchmark, known as GTOC1, models a complex space mission involving multi-gravity assists to asteroid TW229. The objective of this mission is to maximize the change in the semi-major axis of the asteroid's orbit. This benchmark incorporates 8 decision variables and includes 4 constraints that set limits on the proximity to the center during each of the four flyby maneuvers. The variables description includes $var_1$ and $var_2$.

### A.1.6  GTOPX 6: ROSETTA

The sixth benchmark, named Rosetta, emulates multi-gravity-assisted space missions to Comet 67P/Churyumov-Gerasimenko, including the execution of a DSM. The primary objective of this benchmark is to minimize the total $\Delta V$ required throughout the mission. It encompasses 22 decision variables, and description of variables are shown in Table 4.

### A.1.7  GTOPX 7: CASSINI1-MINLP

The last benchmark, cassini1-minLP, is a mixed integer extension of a Cassini1 instance. While in the original Cassini1 instance, the order of planetary flybys was fixed as Venus-Venus-Earth-Jupiter, Cassini1-MINLP treated all four flybys as discrete decision variables. Every planet in the solar system (plus the dwarf planet Pluto) is a viable option to fly by any of the four planets. The benchmark involves 12 decision variables and further considers four constraints. Descriptions of all variables are in Table 4.

## A.2  CEC TASK: INDUSTRIAL AND DESIGN PROBLEMS

For detailed information and access to the resources associated with the task, please visit the provided URL: https://github.com/P-N-Suganthan/2020-RW-Constrained-Optimization. This link leads to the GitHub repository, where you can find all the necessary files, including source code and documentation, to understand and engage with the benchmark suite effectively.

### A.2.1  CEC 1: OPTIMAL OPERATION OF ALKYLATION UNIT

The initial benchmark test is termed "Optimal Operation of Alkylation Unit". This test focuses on optimizing the octane number of the olefin feed in an acidic environment, with the main goal to enhance the alkylating product. The benchmark involves 7 decision variables and incorporates 14 constraints, which are designed to limit onboard fuel and launcher performance. The problem can be formulated as follows.

Minimize:

$f(\bar{x}) = -(0.035x_1x_6 + 1.715x_1 + 10.0x_2 + 4.0565x_3 - 0.063x_3x_5)$

subject to:

$g_1(\bar{x}) = 0.0059553571x_6^2x_1 + 0.88392857x_3 - 0.1175625x_6x_1 - x_1 \leq 0,$

$g_2(\bar{x}) = 1.1088x_1 + 0.1303533x_1x_6 - 0.0066033x_1x_6^2 - x_3 \leq 0,$

$g_3(\bar{x}) = 6.66173269x_6^2 - 56.596669x_4 + 172.39878x_5 - 10000 - 191.20592x_6 \leq 0$

$g_4(\bar{x}) = 1.08702x_6 - 0.03762x_6^2 + 0.32175x_4 + 56.85075 - x_5 \leq 0,$

$g_5(\bar{x}) = 0.006198x_7x_4x_3 + 2462.3121x_225.125634x_2x_4 - x_3x_4 \leq 0,$

$g_6(\bar{x}) = 161.18996x_3x_4 + 5000.0x_2x_4 - 489510.0x_2 - x_3x_4x_7 \leq 0,$

$g_7(\bar{x}) = 0.33x_7 + 44.333333 - x_5 \leq 0,$

$g_8(\bar{x}) = 0.022556x_5 - 1.0 - 0.007595x_7 \leq 0$ (5)

$g_9(\bar{x}) = 0.00061x_3 - 1.0 - 0.0005x_1 \leq 0,$

$g_{10}(\bar{x}) = 0.819672x_1 - x_3 + 0.819672 \leq 0,$

$g_{11}(\bar{x}) = 24599.9x_2250x_2x_4 - x_3x_4 \leq 0,$

$g_{12}(\bar{x}) = 1020.4082x_4x_2 + 1.2244898x_3x_4 - 100000x_2 \leq 0,$

$g_{13}(\bar{x}) = 6.25x_1x_6 + 6.25x_1 - 7.625x_3 - 100000 \leq 0,$

$g_{14}(\bar{x}) = 1.22x_3 - x_6x_1 - x_1 + 1.0 \leq 0.$

with bounds:

$1000 \leq x_1 \leq 2000, 0 \leq x_2 \leq 100,$

$2000 \leq x_3 \leq 4000, 0 \leq x_4 \leq 100,$

$0 \leq x_5 \leq 100, 0 \leq x_6 \leq 20,$

$0 \leq x_7 \leq 200.$

### A.2.2 CEC 2: PROCESS FLOW SHEETING PROBLEM

The second benchmark is characterized as a non-convex constrained optimization problem. This benchmark incorporates three decision variables and is governed by three constraints. It is noted for having an optimal objective function value of $f(x) = 1.07654$. The formulation of this problem can be shown as follows.

Minimize:

$f(\bar{x}) = -0.7x_3 + 0.8 + 5(0.5 - x_1)^2$

subject to:

$g_1(\bar{x}) = -\exp(x_1 - 0.2) - x_2 \leq 0,$

$g_2(\bar{x}) = x_2 + 1.1x_3 \leq -1.0,$ (6)

$g_3(\bar{x}) = x_1 - x_3 \leq 0.2.$

with bounds:

$2.22554 \leq x_2 \leq -1, 0.2 \leq x_1 \leq 1, x_3 \in \{0,1\}.$

### A.2.3 CEC 3: PROCESS SYNTHESIS PROBLEM

The third benchmark, referred to as the Process Synthesis Problem, is defined as a non-convex constrained optimization problem. This benchmark features two decision variables and is subject to two constraints. It is recognized for achieving an best objective function value of $f(x) = 2.0$. The

problem can be defined as follows.

Minimize:

$f(\bar{x}) = x_2 + 2x_1$

subject to:

$g_1(\bar{x}) = -x_1^2 - x_2 + 1.25 \leq 0,$

$g_2(\bar{x}) = x_1 + x_2 \leq 1.6.$

with bounds:

$0 \leq x_1 \leq 1.6,$

$x_2 \in \{0, 1\}.$

$\qquad(7)$

### A.2.4 CEC 4: THREE-BAR TRUSS DESIGN PROBLEM

The fourth benchmark, known as the Three-bar Truss Design Problem, originates from the field of civil engineering and involves a complex constrainted setup. The primary objective of this problem is to reduce the weight of the truss structure. The constraints are based on the stress limits for each bar, leading to a problem characterized by a linear objective function, two decision variables, and three nonlinear constraints. This benchmark's optimal objective function value is $f(x) = 2.63896$. The problem can be defined as below.

Minimize:

$f(\bar{x}) = l\left(x_2 + 2\sqrt{2}x_1\right)$

subject to:

$g_1(\bar{x}) = \dfrac{x_2}{2x_2x_1 + \sqrt{2}x_1^2}P - \sigma \leq 0,$

$g_2(\bar{x}) = \dfrac{x_2 + \sqrt{2}x_1}{2x_2x_1 + \sqrt{2}x_1^2}P - \sigma \leq 0,$

$g_3(\bar{x}) = \dfrac{1}{x_1 + \sqrt{2}x_2}P - \sigma \leq 0.$

where,

$l = 100, P = 2,$ and $\sigma = 2.$

with bounds:

$0 \leq x_1, x_2 \leq 1.$

$\qquad(8)$

### A.2.5 CEC 5: WELDED BEAM DESIGN

The fifth benchmark, known as the Welded Beam Design, primarily aims to minimize the cost of constructing welded beams. This engineering challenge is defined by four variables and five constraints, which guide the development of the beam. The most acclaimed outcome of this benchmark is recorded with an best objective function value of $f(x) = 1.67022$. The description of this problem

is shown below.

Minimize:

$f(\bar{x}) = 0.04811 x_3 x_4 (x_2 + 14) + 1.10471 x_1^2 x_2$

subject to:

$g_1(\bar{x}) = x_1 - x_4 \leq 0,$

$g_2(\bar{x}) = \delta(\bar{x}) - \delta_{\max} \leq 0,$

$g_3(\bar{x}) = P \leq P_c(\bar{x}),$

$g_4(\bar{x}) = \tau_{\max} \geq \tau(\bar{x}),$

$g_5(\bar{x}) = \sigma(\bar{x}) - \sigma_{\max} \leq 0.$

where,

$$\tau = \sqrt{\tau'^2 + \tau''^2 + 2\tau'\tau'' \frac{x_2}{2R}}, \tau'' = \frac{RM}{J}, \tau' = \frac{P}{\sqrt{2} x_2 x_1},$$

$$M = p\left(\frac{x_2}{2} + L\right), \tag{9}$$

$$R = \sqrt{\frac{x_2^2}{4} + \left(\frac{x_1 + x_{xw3}}{2}\right)^2}, J = 2\left(\left(\frac{x_2^2}{4} + \left(\frac{x_1 + x_3}{2}\right)^2\right)\sqrt{2} x_1 x_2\right),$$

$$\sigma(\bar{x}) = \frac{6PL}{x_4 x_3^2}, \delta(\bar{x}) = \frac{6PL^3}{E x_3^2 X_4}, P_c(\bar{x}) = \frac{4.013 E x_3 x_4^3}{6L^2}\left(1 - \frac{x_3}{2L}\sqrt{\frac{E}{4G}}\right)$$

$L = 14\text{in}, P = 6000\text{lb}, E = 30.10^6\text{psi}, \sigma_{\max} = 30,000\text{psi}, \tau_{\max} = 13,600\text{psi},$

$G = 12.10^6\text{psi}, \delta_{\max} = 0.25\text{in}$

with bounds:

$0.1 \leq x_3, x_2 \leq 10$

$0.1 \leq x_4 \leq 2$

$0.125 \leq x_1 \leq 2.$

### A.3 PROTEIN: DNA SEQUENCE OPTIMIZATION

The tasks TF Bind 8 and TF Bind 10 explore the affinity of various DNA sequences for several human transcription factors. In our study, we specifically target the transcription factor SIX6 REF R1, with the aim to design an nucleotide sequence that exhibits high binding affinity to this factor. The datasets for TF Bind 8 and TF Bind 10 encompass all possible combinations of nucleotides for sequences of lengths 8 and 10, respectively, offering a comprehensive basis for assessing sequence effectiveness. You can access the data and code by visiting the URL `https://github.com/brandontrabucco/design-bench`.

## B EXPERIMENTAL DETAILS

In this section, we provide further details about the experiments, including the URL for the code of the comparison algorithms, and the computational resources used.

### B.1 METHODS

This section introduces approaches for offline optimization. To establish a baseline for future comparisons, we benchmark several recent offline MBO algorithms across our tasks. Existing methods fall into three main categories: forward, generative, and evolutionary methods. Forward methods focus on training robust surrogate models to combat adversarial optimization of inputs, followed by gradient-based maximization. Generative methods sample solutions from learned generative models with regularization. Evolutionary methods use neural networks and other techniques to learn proxy models, which are then solved using evolutionary algorithms. In addition, we also introduce traditional black box optimization methods for comparison. For evaluation, we consider three key

components of offline optimization: model architecture, learning algorithm, and search algorithm, as outlined below.

### B.1.1 TRADITIONAL BLACK BOX OPTIMIZATION METHODS

**BO-qEI** Balandat et al. (2019): We perform offline Bayesian optimization by fitting a Gaussian Process using a learned surrogate model $\hat{f}(\boldsymbol{x})$. Next, we employ the quasi-Expected Improvement acquisition function to propose candidate solutions for efficiency. After the Bayesian optimization cycle is completed, we select the best candidates $\boldsymbol{x}$ from the dataset $\hat{\mathcal{D}}$. The procedure of BO-qEI is shown in Algorithm 1. We use the implementation from `https://github.com/brandontrabucco/design-baselines/tree/master/design_baselines/bo_qei`.

**CMA-ES** Hansen (2006): We perform offline optimization using the Covariance Matrix Adaptation Evolution Strategy (CMA-ES), which adapts the covariance matrix of a multivariate normal distribution to guide the search. At each iteration, candidate solutions are sampled from the distribution, and the distribution parameters are updated based on selected high-performing solutions. After completing the optimization cycle, we select the best candidates $\boldsymbol{x}$ from the dataset $\mathcal{D}$. The procedure of CMA-ES is shown in Algorithm 1. We use the implementation from `https://github.com/brandontrabucco/design-baselines/tree/master/design_baselines/cbas`.

### B.1.2 FORWARD METHODS

**ARCOO** Lu et al. (2023): ARCOO learns both a surrogate model and an energy-based model, which characterizes the risk of degradation to address the out-of-distribution issue. The optimizer at each step is regulated by a risk suppression factor derived from the energy-based model. The procedure is outlined in Algorithm 3. The implementation can be found at `https://github.com/luhuakang/ARCOO`.

**Tri-mentoring** Chen et al. (2023): Tri-mentoring trains three surrogate models on the offline dataset and utilizes majority voting to generate consensus labels. To mitigate potential errors in the consensus, an adaptive soft-labeling module is applied. The complete algorithm is presented in Algorithm 4. The implementation can be found at `https://github.com/GGchen1997/parallel_mentoring`.

### B.1.3 GENERATIVE METHODS

**Autofocusing CbAS** Brookes et al. (2019): The Autofocusing CbAS iteratively updates a search model for optimal design by focusing on high-quality solutions. It employs importance sampling to weight samples based on their likelihood of meeting desired conditions, allowing for targeted optimization. Using weighted Maximum Likelihood Estimation (MLE), the search model is refined, while an oracle model is retrained to adaptively reduce bias as the design space is explored. The procedure of Autofocusing CbAS is shown in Algorithm 5. We use the implementation from `https://github.com/brandontrabucco/design-baselines/tree/master/design_baselines/autofocused_cbas`.

---

**Algorithm 1** Offline Bayesian Optimization Trabucco et al. (2022)

---

1: Train a surrogate model $\hat{f}$ based on offline dataset $\mathcal{D}$.
2: Select the top 1% initial designs $\hat{\mathcal{D}} = (\boldsymbol{X}_t, \boldsymbol{y}_t)$ from the offline dataset $\mathcal{D}$.
3: **for** $t = 1...K$ **do**
4:     Find $\boldsymbol{x}_t$ by optimizing the quasi-Expected-Improvemnt acquation function over the Gaussian Process: $\boldsymbol{x}_t = \arg\max_{\boldsymbol{x}} u(\boldsymbol{x} \mid \hat{\mathcal{D}}_{1:t-1})$.
5:     Sample the Surrogate function: $y_t = \hat{f}(\boldsymbol{x}_t)$.
6:     Augment the data $\hat{\mathcal{D}}_{1:t} = \{\hat{\mathcal{D}}_{1:t-1}, (\boldsymbol{x}_t, y_t)\}$ and update the Gaussian Process.
7: **end for**

---

---

**Algorithm 2** Covariance Matrix Adaptation Evolution Strategy (CMA-ES) Hansen (2006)

---

1: Initialize mean $\boldsymbol{m}_0$, step size $\sigma_0$, population size $\lambda$, initial covariance matrix $\boldsymbol{C}_0 = \boldsymbol{I}$
2: Set learning rates for mean ($\mu$), covariance matrix ($\alpha$), and step size adaptation ($\beta$)
3: **for** each generation $t = 0, 1, 2, \ldots$ **do**
4:     Sample $\lambda$ candidate solutions $\boldsymbol{x}_i^{(t)} \sim \mathcal{N}(\boldsymbol{m}_t, \sigma_t^2 \boldsymbol{C}_t)$ for $i = 1, \ldots, \lambda$
5:     Evaluate fitness $f(\boldsymbol{x}_i^{(t)})$ for each candidate $\boldsymbol{x}_i^{(t)}$
6:     Select the top $\mu$ candidates with the best fitness values
7:     Update mean $\boldsymbol{m}_{t+1}$ as the weighted average of the top $\mu$ candidates:

$$\boldsymbol{m}_{t+1} = \sum_{i=1}^{\mu} w_i \boldsymbol{x}_i^{(t)}$$

8:     Update covariance matrix $\boldsymbol{C}_{t+1}$ to adapt the search distribution:

$$\boldsymbol{C}_{t+1} = (1 - \alpha)\boldsymbol{C}_t + \alpha \sum_{i=1}^{\mu} w_i \left( \boldsymbol{x}_i^{(t)} - \boldsymbol{m}_t \right) \left( \boldsymbol{x}_i^{(t)} - \boldsymbol{m}_t \right)^T$$

9:     Update step size $\sigma_{t+1}$ based on the success rate of candidate solutions:

$$\sigma_{t+1} = \sigma_t \exp \left( \beta \left( \frac{\|\boldsymbol{p}_\sigma\|}{\|\mathcal{N}(0, \boldsymbol{I})\|} - 1 \right) \right)$$

10:     Update evolution path $\boldsymbol{p}_\sigma$ for step size adaptation
11: **end for**
12: **return** Best solutions found during the optimization process

---

### B.1.4 EVOLUTIONARY METHODS

**TTDDEA** Huang et al. (2021): TTDDEA is an offline optimization method that combines tri-training with evolutionary algorithms. It uses three Radial Basis Function Networks (RBFNs) as surrogate models to predict fitness scores and generates high-confidence pseudo-labels to augment limited training data. By applying an evolutionary optimization process, TT-DDEA iteratively updates these surrogate models and selects high-performing candidate solutions for the next generation. This approach leverages semi-supervised learning and multi-model ensembles to address data scarcity in offline environments. The algorithm can be seen in Algorithm 6. We use the implementation from `https://github.com/HandingWangXDGroup/TT-DDEA`.

**CCDDEA** Gong et al. (2023): CCDDEA combines hierarchical surrogate models with cooperative coevolution to solve large-scale optimization problems. It uses a global model (HM) and local models (LMs) to guide search at different levels. Local searches are enhanced with gradient-based and evolutionary operators, while global communication merges sub-populations. The dynamic space division strategy improves convergence by shifting focus from local to global optimization. The procedure of CCDDEA is shown in Algorithm. The implementation can be found in `https://github.com/LabGong/cc-ddea`.

### B.1.5 CONSTRAINED METHODS

We select two primary categories of offline constraint algorithms: deep learning-based constrained offline methods and evolution-based constrained offline methods. For our experiments, we utilize classic methods from both categories.

**CARCOO**: This is a simplified constrained version of ARCOO. We integrate the degree of constraint violation into risk assessment, treating solutions that violate constraints as high risk. CARCOO employs the three models (i.e., a surrogate model, an energy-based model to characterize OOD risk, an energy-based model to characterize constrained risk) to train the models on a training dataset. The algorithm can be found in Algorithm 8.

---

**Algorithm 3** Accumulative Risk Controlled Offline Optimization (ARCOO) Lu et al. (2023)

---

**Input**:Offline dataset $\mathcal{D}$, learning rate $\eta$, maximum Langevin dynamics step $K$, Langevin dynamics stepsize $\lambda$, and initial momentum $m$.

1: Initialize dual-head model that consists of surrogate head $\hat{f}_\theta(\boldsymbol{x})$ and energy head $E_\phi(\boldsymbol{x})$.
2: **for** each training epoch **do**
3:     Update $\hat{f}_\theta(\boldsymbol{x})$ using MSE loss:
$$\theta \leftarrow \theta - \eta\nabla_\theta\mathcal{L}_\mathcal{D}(\theta)$$

4:     Sample high-risk distribution $q(\boldsymbol{x})$ by Langevin dynamics: $q(\boldsymbol{x}) = \mathrm{LD}_\theta(p(\boldsymbol{x}); K)$, i.e.,
$$\boldsymbol{x}_k \leftarrow \boldsymbol{x}_{k-1} + \lambda\nabla_{\boldsymbol{x}}\hat{f}_\theta(\boldsymbol{x}_{k-1}) + \omega_k, \quad k = 1, \ldots, K,$$
    where $\omega_k \sim \mathcal{N}(0, \lambda)$, and $\boldsymbol{x}_0 \sim p(\boldsymbol{x})$. Sampling starts from the low-risk empirical distribution $p(\boldsymbol{x})$ over the offline dataset.
5:     Update $E_\phi(\boldsymbol{x})$ using contrastive divergence loss:
$$\phi \leftarrow \phi - \eta\nabla_\phi\left[\mathrm{KL}(p(\boldsymbol{x})\|h_\phi(\boldsymbol{x})) - \mathrm{KL}(q(\boldsymbol{x})\|h_\phi(\boldsymbol{x}))\right],$$
    where $h_\phi$ is derived from $E_\phi(\boldsymbol{x})$.
6: **end for**
7: Let $\tilde{\mathcal{P}}$ be an empirical distribution over a batch of the high-quality solutions in $\mathcal{D}$, and $\tilde{\mathcal{Q}} = \mathrm{LD}_\theta(\tilde{\mathcal{P}}; K)$.
8: Calculate the risk suppression factor:
$$R_\phi(\boldsymbol{x}) = m(E_{\tilde{\mathcal{Q}}} - E_\phi(\boldsymbol{x}))(E_{\tilde{\mathcal{Q}}} - E_{\tilde{\mathcal{P}}})^{-1}.$$

9: **for** $t = 1$ to $T$ **do**
10:     $\boldsymbol{x}_t \leftarrow \boldsymbol{x}_{t-1} + R_\phi(\boldsymbol{x}_{t-1})\nabla_{\boldsymbol{x}}\hat{f}_\theta(\boldsymbol{x}_{t-1})$.
11: **end for**
12: **return** Final solution $\boldsymbol{x}_{\mathrm{app}} = \boldsymbol{x}_T$ for online application.

---

**Algorithm 4** Tri-mentoring for Offline Model-based Optimization Chen et al. (2023)

---

**Input**: The static dataset $\mathcal{D}$, the number of iterations $T$, the optimizer $\mathrm{OPT}(\cdot)$.

1: Initialize $x_0$ as the design with the highest score in $\mathcal{D}$.
2: Train proxies $f_\theta^A(\cdot)$, $f_\theta^B(\cdot)$, and $f_\theta^C(\cdot)$ on $\mathcal{D}$ with different initializations.
3: **for** $t \leftarrow 0$ to $T - 1$ **do**
4:     Sample $K$ neighborhood points at $x_t$ as $S(\boldsymbol{x}_t)$.
5:     Compute pairwise comparison labels $\hat{y}^A$, $\hat{y}^B$, and $\hat{y}^C$ for the three proxies on $S(\boldsymbol{x}_t)$.
6:     Derive consensus labels: $\hat{y}^V = \mathrm{majority\_voting}(\hat{y}^A, \hat{y}^B, \hat{y}^C)$.
7:     **for** proxy in $\{f_\theta^A(\cdot), f_\theta^B(\cdot), f_\theta^C(\cdot)\}$ **do**
8:         Initialize soft-labels as consensus labels: $\hat{y}^S = \hat{y}^V$.
9:         *Inner level*: fine-tune the proxy with Eq. (8).
10:         *Outer level*: learn more accurate soft-labels $\hat{y}^S$ with Eq. (9).
11:         Mentor proxy using the optimized soft-labels $\hat{y}^S$ with Eq. (8).
12:     **end for**
13:     Form a more robust ensemble as $f_\theta(x) = \frac{1}{3}\left(f_\theta^A(\boldsymbol{x}) + f_\theta^B(\boldsymbol{x}) + f_\theta^C(\boldsymbol{x})\right)$.
14:     Gradient ascent: $\boldsymbol{x}_{t+1} = \boldsymbol{x}_t + \eta\mathrm{OPT}(\nabla_{\boldsymbol{x}}f_\theta(\boldsymbol{x}_t))$.
15: **end for**
16: **return** The high-scoring designs $\boldsymbol{x}^* = \boldsymbol{x}_T$.

---

**DDEA-PF & DDEA-SPF** Huang & Wang (2021b): DDEA-PF and DDEA-SPF are two data-driven evolutionary algorithms for handling constraints in optimization tasks. DDEA-PF employs penalty functions to manage constraints, adjusting the penalty to push solutions towards feasibility. On the other hand, DDEA-SPF focuses on adaptive penalty, i.e., $\mathrm{fitness}(\boldsymbol{x}) = f(\boldsymbol{x}) + r_i\frac{(\sum_{j=1}^M \max[0, g_j(\boldsymbol{x})]^q)}{g_{\mathrm{max}, j}}$.

---

**Algorithm 5** Autofocused Model-based Optimization (Autofocusing CbAS) Brookes et al. (2019)

---

**Input**: Offline dataset, $\mathcal{D} = \{(\boldsymbol{x}_i, y_i)\}_{i=1}^n$; oracle model class, $p_\beta(y \mid \boldsymbol{x})$ with parameters, $\beta$, that can be estimated with MLE; search model class, $p_\theta(\boldsymbol{x})$ with parameters, $\theta$, that can be estimated with weighted MLE or approximations thereof; desired constraint set, $S$ (e.g., $S = \{y \mid y \geq y_\tau\}$); maximum number of iterations, $T$; number of samples to generate, $m$; EDA-specific monotonic transformation, $V(\cdot)$. **Initialization**: Obtain $p_\theta(\boldsymbol{x})$ by fitting to $\{\boldsymbol{x}_i\}_{i=1}^n$ with the search model class. For the search model, set $p_{\theta(0)}(\boldsymbol{x}) \leftarrow p_\theta(\boldsymbol{x})$. For the oracle, $p_{\beta(0)}(y \mid \boldsymbol{x})$, use MLE with equally weighted training data.

1: **for** $t = 1, \ldots, T$ **do**
2:      Sample from the current search model, $\{\tilde{\boldsymbol{x}}_i^{(t)}\}_{i=1}^m \sim p_{\theta(t-1)}(\boldsymbol{x})$, $\forall i \in \{1, \ldots, m\}$.
3:      $v_i \leftarrow V\left(P_{\beta(t-1)}(y \in S \mid \tilde{\boldsymbol{x}}_i^{(t)})\right)$, $\forall i \in \{1, \ldots, m\}$.
4:      Fit the updated search model, $p_{\theta(t)}(\boldsymbol{x})$, using weighted MLE with the samples, $\{\tilde{\boldsymbol{x}}_i^{(t)}\}_{i=1}^m$, and their corresponding EDA weights, $\{v_i\}_{i=1}^m$.
5:      Compute importance weights for the training data, $w_i \leftarrow p_{\theta(t)}(\boldsymbol{x}_i)/p_{\theta(0)}(\boldsymbol{x}_i)$, $i = 1, \ldots, n$.
6:      Retrain the oracle using the re-weighted training data,

$$\beta(t) \leftarrow \arg\max_{\beta \in B} \frac{1}{n} \sum_{i=1}^n w_i \log p_\beta(y_i \mid \boldsymbol{x}_i).$$

7: **end for**
8: **return** The most promising candidates among $\{\tilde{\boldsymbol{x}}_i^{(t)}, ..., \tilde{\boldsymbol{x}}_m^{(t)}\}_{t=1}^T$.

---

**Algorithm 6** Tri-Training Data-Driven Evolutionary Algorithm (TTDDEA) Huang et al. (2021)

---

**Input**: Separate offline data sets $L_1$, $L_2$, and $L_3$, trained RBFNs $M_1$, $M_2$, and $M_3$, current population $P$, population size $Q$.

1: **for** $i = 1 \rightarrow 3$ **do**
2:      Use the models $M_1$, $M_2$, $M_3$ to predict the fitness of population, written as $f_1^1, f_2^1, \ldots, f_Q^1$, $f_1^2, f_2^2, \ldots, f_Q^2$, and $f_1^3, f_2^3, \ldots, f_Q^3$.
3: **end for**
4: **for** $i = 1 \rightarrow 3$ **do**
5:      Find the high-confidence data $\boldsymbol{x}_i$ as in Equation (9).
6:      Calculate its pseudo label $\hat{y}_i$ as in Equation (10).
7:      Update $L_i$ to $L_i'$ by $(\boldsymbol{x}_i, \hat{y}_i)$ as in Equation (11).
8: **end for**
9: Train RBFNs $M_1$, $M_2$, and $M_3$ with $L_1'$, $L_2'$, and $L_3'$.
10: Use the evolutionary algorithms to obtain promising candidates.
11: **return** The most promising candidates found by the evolutionary algorithms.

---

DDEA-PF is presented in Algorithm 9. The implementation is from `https://github.com/HandingWangXDGroup/Constraint-Handling-OfflineDDEA`.

**CCOMs**: This is a simple experimental version of PRIME (Kumar et al., 2022). We use $\theta^* = \arg\min_\theta \mathbb{E}_{\boldsymbol{x}_i, y_i \sim \mathcal{D}_{\text{feasible}}} \left[(f_\theta(\boldsymbol{x}_i) - y_i)^2\right] - \alpha \mathbb{E}_{\boldsymbol{x}_i^- \sim \mathcal{D}_{\text{infeasible}}} \left[f_\theta(\boldsymbol{x}_i^-)\right]$. This means that in addition to fitting the surrogate model, we also make the values of points that violate the constraints as small as possible. Then use the conservative model COMs (Trabucco et al., 2021) as surrogate model to train. Pseudocode can be found at Algorithm 10.

### B.2 COMPUTATION RESOURCES

The computing resources required for the research described in this paper are relatively modest, requiring only a single Nvidia GeForce RTX 3090 GPU. The experiments were efficiently completed using this powerful graphics card, almost all within a 24-hour timeframe.

---

**Algorithm 7** Offline Data-Driven Optimization at Scale: A Cooperative Coevolutionary Approach (CCDDEA) Gong et al. (2023)

---

1: **Input:** $\mathcal{D}$: The offline data;
        $n$: The size of complete population;
        $T_H, T_L$: The maximum generations of higher-level and lower-level optimization;
        $g_{\text{init}}$: The initial number of groups;
        $T_g$: The interval for updating the number of groups;
        $T_r$: The interval for re-dividing the sub-spaces;
        $\alpha$: The learning rate of gradient descent;
        $r_{\text{top}}$: The control parameter for top-ranked random merging in the cooperative search;
2: **Output:** The best solution
3: **Initialization**: $i_r \leftarrow 0$ (index of $T_r$), $g \leftarrow g_{\text{init}}$
4: $P \leftarrow$ Latin hypercube sampling
5: **for** $i \leftarrow 1$ to $T_H$ **do**
6:    **if** $i > 1$ and $i \bmod T_g = 1$ and $g > 1$ **then**
7:       $g \leftarrow g - 1$
8:       $i_r \leftarrow 0$
9:    **end if**
10:   **if** $i_r \bmod T_r = 0$ and $g > 1$ **then**
11:      $G \leftarrow$ division rules
12:      $(HM, LM_i) \leftarrow$ HSJL$(\mathcal{D}, G)$
13:   **end if**
14:   $SP \leftarrow$ split $P$ according to $G$
15:   **for** $SP_j \in SP$ **do**
16:      $SP_j \leftarrow$ LowerLevelSearch$(SP_j, T_L, \alpha, LM_j)$
17:   **end for**
18:   **if** $g > 1$ **then**
19:      $P \leftarrow$ HigherLevelSearch$(SP, r_{top}, HM, LMs)$
20:   **end if**
21:   $i_r \leftarrow i_r + 1$
22: **end for**
23: **return** $P[0]$

---

## C EXPERIMENTAL RESULTS

Here, we present detailed results across various tasks. For each task, we evaluate 128 solutions at the 100th percentiles. In this paper, we define our areas (i.e., $S, S_1, S_2$) as the region formed by doubling the distance between solutions (i.e., the optimal algorithm for each step in the optimization process is in $100_{\text{th}}S$, the optimal offline solution in $S_1$, and the optimal solution found by the algorithm in $S_2$) and algorithm's optimal solution and relative to the offline optimal solution, as shown in Eq. (2). Additionally, we assess the algorithms' robustness by providing results at the 50th and 0th percentiles. Among the various algorithms, ARCOO, CMAES, and Trimentoring emerge as the top performers. However, most algorithms experience a decline in performance as the number of optimization steps increases from 100 to 150. This highlights the necessity for further advancements in the stability of offline black-box optimization methods, particularly for addressing these challenging tasks.

Due to the significant computational cost associated with the CMAES algorithm, which requires more than 24 hours to complete, this paper focuses on reporting the results after only 50 optimization steps.

The experimental results in the unconstrained scenario show that ARCOO and CMA-ES algorithms are the most balanced and stable on various benchmark functions, demonstrating high performance and stability. Although CCDDEA performs well in some cases, the stability of the results is poor. This may be because previous offline evolutionary algorithms did not consider the problem of out-of-distribution generalization.

Since CbAS is a generative model that samples the design space and optimizes based on the conditional distribution of the target attribute, the final result is primarily determined by the model's

---

**Algorithm 8** Constrained Accumulative Risk Controlled Offline Optimization (CARCOO)

---

**Input**:Offline dataset $\mathcal{D}$, learning rate $\eta$, maximum Langevin dynamics step $K$, Langevin dynamics stepsize $\lambda$, and initial momentum $m$.

1: Initialize dual-head model that consists of surrogate head $\hat{f}_\theta(\boldsymbol{x})$ and energy head $E_\phi(\boldsymbol{x})$.
2: **for** each training epoch **do**
3:   Update $\hat{f}_\theta(\boldsymbol{x})$ using MSE loss:
$$\theta \leftarrow \theta - \eta \nabla_\theta \mathcal{L}_\mathcal{D}(\theta)$$

4:   Sample high-risk distribution $q(\boldsymbol{x})$ by Langevin dynamics: $q(\boldsymbol{x}) = \mathrm{LD}_\theta(p(\boldsymbol{x}); K)$, i.e.,
$$\boldsymbol{x}_k \leftarrow \boldsymbol{x}_{k-1} + \lambda \nabla_{\boldsymbol{x}} \hat{f}_\theta(\boldsymbol{x}_{k-1}) + \omega_k, \quad k = 1, \ldots, K,$$
where $\omega_k \sim \mathcal{N}(0, \lambda)$, and $\boldsymbol{x}_0 \sim p(\boldsymbol{x})$. Sampling starts from the low-risk empirical distribution $p(x)$ over the offline dataset.
5:   Update $E_\phi(\boldsymbol{x})$ using contrastive divergence loss:
$$\phi \leftarrow \phi - \eta \nabla_\phi \left[ \mathrm{KL}(p(\boldsymbol{x}) \| h_\phi(\boldsymbol{x})) - \mathrm{KL}(q(\boldsymbol{x}) \| h_\phi(\boldsymbol{x})) \right],$$
where $h_\phi$ is derived from $E_\phi(\boldsymbol{x})$.
6:   Update $\hat{E}_\tau(\boldsymbol{x})$ using contrastive divergence loss:
$$\tau \leftarrow \tau - \eta \nabla_\tau \left[ \mathrm{KL}(\hat{p}(\boldsymbol{x}) \| \hat{h}_\tau(\boldsymbol{x})) - \mathrm{KL}(\hat{q}(\boldsymbol{x}) \| \hat{h}_\tau(\boldsymbol{x})) \right],$$
where $\hat{h}_\tau$ is derived from $\hat{E}_\phi(\boldsymbol{x})$, $\hat{p}(\boldsymbol{x})$ is solutions that satisfy the constraints in the dataset, $\hat{q}(\boldsymbol{x})$ is solutions in the dataset that do not satisfy the constraints
7: **end for**
8: Let $\tilde{\mathcal{P}}$ be an empirical distribution over a batch of the high-quality solutions in $\mathcal{D}$, and $\tilde{\mathcal{Q}} = \mathrm{LD}_\theta(\tilde{\mathcal{P}}; K)$.
9: Calculate the risk suppression factor:
$$R_\phi(\boldsymbol{x}) = m(E_{\tilde{\mathcal{Q}}} - E_\phi(\boldsymbol{x}))(E_{\tilde{\mathcal{Q}}} - E_{\tilde{\mathcal{P}}})^{-1}.$$
$$R_\tau(\boldsymbol{x}) = m(E_{\tilde{\mathcal{Q}}} - E_\tau(\boldsymbol{x}))(E_{\tilde{\mathcal{Q}}} - E_{\tilde{\mathcal{P}}})^{-1}.$$

10: **for** $t = 1$ to $T$ **do**
11:   $\boldsymbol{x}_t \leftarrow \boldsymbol{x}_{t-1} + \frac{(R_\phi(\boldsymbol{x}_{t-1}) + R_\tau(\boldsymbol{x}_{t-1}))}{2} \nabla_{\boldsymbol{x}} \hat{f}_\theta(\boldsymbol{x}_{t-1})$.
12: **end for**
13: **return** Final solution $\boldsymbol{x}_{\mathrm{app}} = \boldsymbol{x}_T$ for online application.

---

conditional distribution and the adaptive sampling strategy, rather than the step size. When the seed is fixed, the result is derived from the conditional distribution regardless of the size of the optimization step, leading to consistent outcomes. Furthermore, due to the inherent complexity of satellite missions, the VAE model struggles to effectively learn the prior distribution, preventing it from surpassing the offline optimal.

The experimental results in the constraint scenario show that the DE-PF and DE-SFP algorithms do not perform well on various benchmark functions. In addition, the performance of DE-PF and DE-SFP on multiple benchmark functions is almost the same, showing consistent performance. The performance of the CCOMS algorithm fluctuates greatly in different dimensions and functions. Overall, the CARCOO algorithm has good performance and stability when dealing with constrained optimization problems, providing an important reference for subsequent algorithm optimization. However, CARCOO and CCOMs often fail to find a suitable feasible solution due to the small number of data points and insufficient neural network training.

# D  Hyperparameter Analysis

We outline guidelines for hyperparameter selection for methods evaluated in the benchmark. These principles help in offline tuning for new tasks. Begin by identifying essential hyperparameters, like learning rate and batch size, and select a tuning method such as Grid Search or Random Search. We run experiments to find the satisfied hyperparameter configurations with performance metrics like accuracy and F1 score.

## D.1  Hyperparameters of BO-qEI

The primary tunable components of Bayesian optimization include the objective function and the parameters within the optimization loop. The objective function is commonly optimized with a maximum likelihood method, allowing validation log-likelihood or regression error to be tracked directly. Training continues until the validation loss is minimized, ensuring good generalization beyond the training dataset. This tuning method is fully offline since it exclusively uses a static task dataset. In this study, we employ a Gaussian Process model combined with the quasi-Monte Carlo Expected Improvement acquisition function. For an in-depth understanding of various Bayesian optimization strategies and their associated hyperparameters, detailed information can be found in the BoTorch documentation at `https://botorch.org/docs/overview`.

## D.2  Hyperparameters of CMA-ES

The primary configurable elements of Covariance Matrix Adaptation (CMA) techniques include the trained objective function and the evolution strategy parameters. Typically, the objective function is trained using a maximum likelihood approach to ensure optimal performance. For further details and comprehensive information, refer to the open-source CMA-ES implementation and its extensive documentation available at `https://github.com/CMA-ES/pycma`. In this research, we adopt the default settings for CMA-ES provided by this implementation, with the parameter $\sigma$ set to 0.5.

## D.3  Hyperparameters of Autofocusing CbAS

Autofocusing CbAS has one main tunable components. It involves verifying the performance index, which plays a crucial role in generating the generalization capability of the learning model. By ensuring that the performance index stays above a certain positive threshold, we can ascertain that the algorithm is well-tuned and performing optimally. In this paper, we have set this threshold to 0.9. This careful selection helps maintain the reliability and effectiveness of the autofocusing CbAS approach.

---

**Algorithm 9** Data-Driven Evolutionary Optimization with Penalty Function (DDEA-PF) Huang & Wang (2021b)

---

1: Initialize population $P$ with size $Q$
2: Set the degree of violation of the $j$th constraint $g_j(\boldsymbol{x})$, penalty coefficient $r_i$.
3: **for** each generation T **do**
4:   **for** each individual $x$ in $P$ **do**
5:     Evaluate fitness using the objective function with a penalty for constraint violation:

$$\text{fitness}(x) = f(\boldsymbol{x}) + r_i(\sum_{j=1}^{M} \max[0, g_j(\boldsymbol{x})]^q)$$

6:   **end for**
7:   Perform selection, crossover, and mutation in population $P$ to find best feasible solutions.
8: **end for**
9: **return**  Best feasible solutions $\boldsymbol{x}^* = \boldsymbol{x}_T$.

---

---

**Algorithm 10** Constrained Conservative Objective Models for Offline Optimization (CCOMs)

---

**Input**: Offline dataset $\mathcal{D}$, the learning rate of OOD gradient ascent $\eta$, trade-off coefficient $\alpha$.
**Initialization**: Surrogate model $\hat{f}_\theta$.

1: **for** $i = 1$ to training_steps **do**
2:     Sample $(\boldsymbol{x}_0, y) \sim \mathcal{D}$
3:     Find $\boldsymbol{x}_T(x_0)$ via gradient ascent from $\boldsymbol{x}_0$: $\boldsymbol{x}_{t+1} = \boldsymbol{x}_t + \eta \nabla_{\boldsymbol{x}} \hat{f}_\theta(\boldsymbol{x})\big|_{\boldsymbol{x}=\boldsymbol{x}_t}$;   $\mu(\boldsymbol{x}) = \sum_{\boldsymbol{x}_0 \in \mathcal{D}} \delta_{\boldsymbol{x}=\boldsymbol{x}_T(\boldsymbol{x}_0)}$
4:     Minimize $\mathcal{L}(\theta; \alpha)$ with respect to $\theta$.

$$\mathcal{L}(\theta; \alpha) = \mathbb{E}_{\boldsymbol{x}_0 \sim \mathcal{D}} \left( \hat{f}_\theta(\boldsymbol{x}_0) - y \right)^2 - \alpha \mathbb{E}_{\boldsymbol{x}_0} \left[ \hat{f}_\theta(\boldsymbol{x}_0) \right] + \alpha \mathbb{E}_{\mu(\boldsymbol{x})} \left[ \hat{f}_\theta(\boldsymbol{x}) \right] - \mathbb{E}_{\boldsymbol{x}^- \sim \mathcal{D}_{\inf}} \left[ f_\theta \left( \boldsymbol{x}^- \right) \right]$$

$$\theta \leftarrow \theta - \lambda \nabla_\theta \mathcal{L}(\theta; \alpha)$$

5: **end for**
6: Initialize optimizer at the optimum in $\mathcal{D}$:

7: Find $\boldsymbol{x}^\star$ via trust-region gradient ascent from $\tilde{\boldsymbol{x}}$:

$$\boldsymbol{x}_{t+1} = \boldsymbol{x}_t + \eta \nabla_x \mathcal{L}_{\text{opt}}(\boldsymbol{x})\big|_{\boldsymbol{x}=\boldsymbol{x}_t}$$

    where $\mathcal{L}_{\text{opt}}(\boldsymbol{x}) := \hat{f}_\theta^\star(\boldsymbol{x})$.
8: Return the solution $\boldsymbol{x}^\star = \boldsymbol{x}_T$.

---

## D.4 HYPERPARAMETERS OF ARCOO

The Accumulative Risk Controlled Offline Optimization (ARCOO) method has two hyperparameters: accumulative risk control and Langevin dynamics steps. The former represents the level of total risk tolerance, with an initial momentum set at 0.2, reflecting a moderate risk tolerance. The latter involves sampling by 64 steps of Langevin dynamics, determining the distance between high-risk solutions and observed solutions.

## D.5 HYPERPARAMETERS OF TRI-MENTORING

We tuned two critical hyperparameters: the number of neighborhood samples and the number of optimization steps. Due to the algorithm's robustness to the number of neighborhood samples, we set this parameter to 10. To maintain consistency, we standardized the number of optimization steps to 100 across all experiments.

## D.6 HYPERPARAMETERS OF TTDDEA

In this evolutionary offline optimization task, we retained the original hyperparameters because they were carefully tuned for mutation rate, crossover probability, and selection pressure. These settings ensure a balanced exploration and exploitation process, preventing premature convergence and maintaining a diverse solution population. Keeping them unchanged guarantees the algorithm's robustness and reliable performance.

## D.7 HYPERPARAMETERS OF CARCOO

The hyperparameters in the constrained ARCOO (CARCOO) method need to be consistent with the original ARCOO settings. Therefore, we have configured them to exactly match the hyperparameters used in the original ARCOO method.

### D.8 HYPERPARAMETERS OF CCOMS

Constrainted conservative objective models have four main tunable parameters. The first parameter is the degree to which the objective model is allowed to overestimate the objective value for off-manifold designs, set to 2 to ensure a conservative approach. The second parameter is the number of gradient ascent steps, chosen to be 100, balancing optimization efficiency and computational effort. The third parameter is the learning rate, set to $2\sqrt{d}$, where $d$ is the dimension of the design space, ensuring step sizes are appropriately scaled to the problem's complexity. The final parameter is the constraint trade-off coefficient, set to 1, ensuring an equal emphasis on optimizing the objective and satisfying the constraints. These parameters are crucial for fine-tuning the model's performance within the desired constraints.

### D.9 HYPERPARAMETERS OF DE-PF & DE-SPF

This hyperparameter is identical to the one used in TTDDEA, maintaining consistency with the original paper. However, due to the algorithm's suboptimal performance in handling out-of-distribution (OOD) problems, we have adjusted the number of iterations to 50 in this study. This modification aims to improve the algorithm's efficiency and effectiveness in addressing the specific challenges posed by OOD scenarios.

Table 5: Overall results for GTOPX unconstrained tasks with 128 solutions and 100th percentile evaluations are averaged over eight runs. $f(x^*_{\text{OFF}})$ represents the optimal objective function value in the offline dataset. FS (i.e., final score) refers to the function value found by an offline optimization algorithm at the final step of specific optimization steps (i.e., $t = 50, 100, 150$). The symbol "-" means that the algorithm cannot complete the corresponding task within 24 hours. In this case, 0% of the values are missing near the worst value and another 50% near the optimal value.

| Steps | | | t = 50 | | | | | t = 100 | | | | | t = 150 | | |
|---|---|---|---|---|---|---|---|---|---|---|---|---|---|---|---|
| Task | | | | | | | | GTOPX 2 | | | | | | | |
| $f(x^*_{\text{OFF}})$ | | | | | | | | $-356.70_{\pm 2.33}$ | | | | | | | |
| Metric | FS | SI | OI | SO | SO$_w$ | FS | SI | OI | SO | SO$_w$ | FS | SI | OI | SO | SO$_w$ |
| ARCOO | $-59.17_{\pm 9.56}$ | $0.97_{\pm 0.01}$ | $1.93_{\pm 0.02}$ | $1.29_{\pm 0.01}$ | $1.45_{\pm 0.01}$ | $-59.58_{\pm 9.95}$ | $0.98_{\pm 0.01}$ | $1.96_{\pm 0.02}$ | $1.30_{\pm 0.01}$ | $1.18_{\pm 0.01}$ | $-59.49_{\pm 10.61}$ | $0.98_{\pm 0.01}$ | $1.96_{\pm 0.02}$ | $1.31_{\pm 0.02}$ | $0.98_{\pm 0.01}$ |
| BO | $-95.63_{\pm 22.06}$ | $0.93_{\pm 0.01}$ | $1.85_{\pm 0.02}$ | $1.24_{\pm 0.00}$ | $1.39_{\pm 0.00}$ | $-109.73_{\pm 25.18}$ | $0.93_{\pm 0.01}$ | $1.85_{\pm 0.02}$ | $1.23_{\pm 0.01}$ | $1.11_{\pm 0.01}$ | $-88.91_{\pm 12.52}$ | $0.93_{\pm 0.01}$ | $1.85_{\pm 0.01}$ | $1.23_{\pm 0.00}$ | $0.93_{\pm 0.01}$ |
| CBAS | $-356.70_{\pm 2.33}$ | $1.00_{\pm 0.00}$ | $1.00_{\pm 0.00}$ | $1.00_{\pm 0.00}$ | $1.00_{\pm 0.00}$ | $-356.70_{\pm 2.33}$ | $1.00_{\pm 0.00}$ | $1.00_{\pm 0.00}$ | $1.00_{\pm 0.00}$ | $1.00_{\pm 0.00}$ | $-356.70_{\pm 2.33}$ | $1.00_{\pm 0.00}$ | $1.00_{\pm 0.00}$ | $1.00_{\pm 0.00}$ | $1.00_{\pm 0.00}$ |
| CCDDEA | $-152.69_{\pm 27.88}$ | $0.73_{\pm 0.08}$ | $1.46_{\pm 0.16}$ | $0.98_{\pm 0.11}$ | $1.10_{\pm 0.12}$ | $-2.61e3_{\pm 3.71e3}$ | $-0.12_{\pm 1.10}$ | $-0.25_{\pm 2.20}$ | $-0.17_{\pm 1.46}$ | $-0.15_{\pm 1.32}$ | $-3.71e3_{\pm 4.57e3}$ | $-1.67_{\pm 3.03}$ | $-3.331_{\pm 6.050}$ | $-2.22_{\pm 4.03}$ | $-1.67_{\pm 3.04}$ |
| CMAES | $-57.74_{\pm 6.54}$ | $0.98_{\pm 0.01}$ | $1.96_{\pm 0.01}$ | $1.31_{\pm 0.01}$ | $1.47_{\pm 0.01}$ | - | - | - | - | - | - | - | - | - | - |
| TTDDEA | $-301.71_{\pm 226.55}$ | $0.75_{\pm 0.13}$ | $1.50_{\pm 0.26}$ | $1.00_{\pm 0.17}$ | $1.13_{\pm 0.20}$ | $-251.94_{\pm 121.20}$ | $0.705_{\pm 0.20}$ | $1.41_{\pm 0.40}$ | $0.94_{\pm 0.26}$ | $0.85_{\pm 0.24}$ | $-236.05_{\pm 45.77}$ | $0.70_{\pm 0.19}$ | $1.40_{\pm 0.38}$ | $0.93_{\pm 0.25}$ | $0.70_{\pm 0.19}$ |
| Trimentoring | $-53.58_{\pm 5.73}$ | $0.99_{\pm 0.00}$ | $1.97_{\pm 0.01}$ | $1.31_{\pm 0.01}$ | $1.48_{\pm 0.00}$ | $-54.63_{\pm 7.10}$ | $0.99_{\pm 0.00}$ | $1.98_{\pm 0.01}$ | $1.32_{\pm 0.01}$ | $1.19_{\pm 0.00}$ | $-56.43_{\pm 6.58}$ | $0.99_{\pm 0.00}$ | $1.98_{\pm 0.01}$ | $1.32_{\pm 0.01}$ | $0.99_{\pm 0.00}$ |
| Task | | | | | | | | GTOPX 3 | | | | | | | |
| $f(x^*_{\text{OFF}})$ | | | | | | | | $-336.15_{\pm 6.28}$ | | | | | | | |
| ARCOO | $-77.75_{\pm 53.15}$ | $0.96_{\pm 0.03}$ | $1.91_{\pm 0.05}$ | $1.27_{\pm 0.04}$ | $1.43_{\pm 0.04}$ | $-66.60_{\pm 24.91}$ | $0.95_{\pm 0.03}$ | $1.90_{\pm 0.07}$ | $1.27_{\pm 0.05}$ | $1.15_{\pm 0.04}$ | $-82.56_{\pm 45.07}$ | $0.95_{\pm 0.04}$ | $1.90_{\pm 0.08}$ | $1.26_{\pm 0.06}$ | $0.95_{\pm 0.04}$ |
| BO | $-69.00_{\pm 14.86}$ | $0.94_{\pm 0.01}$ | $1.87_{\pm 0.01}$ | $1.25_{\pm 0.01}$ | $1.41_{\pm 0.01}$ | $-58.05_{\pm 14.82}$ | $0.94_{\pm 0.01}$ | $1.88_{\pm 0.02}$ | $1.25_{\pm 0.01}$ | $1.13_{\pm 0.01}$ | $-60.00_{\pm 8.22}$ | $0.94_{\pm 0.01}$ | $1.88_{\pm 0.02}$ | $1.25_{\pm 0.01}$ | $0.94_{\pm 0.01}$ |
| CBAS | $-336.15_{\pm 6.28}$ | $1.00_{\pm 0.00}$ | $1.00_{\pm 0.00}$ | $1.00_{\pm 0.00}$ | $1.00_{\pm 0.00}$ | $-336.15_{\pm 6.28}$ | $1.00_{\pm 0.00}$ | $1.00_{\pm 0.00}$ | $1.00_{\pm 0.00}$ | $1.00_{\pm 0.00}$ | $-336.15_{\pm 6.28}$ | $1.00_{\pm 0.00}$ | $1.00_{\pm 0.00}$ | $1.00_{\pm 0.00}$ | $1.00_{\pm 0.00}$ |
| CCDDEA | $-2.43e4_{\pm 6.32e4}$ | $-21.61_{\pm 44.10}$ | $-43.15_{\pm 88.20}$ | $-28.80_{\pm 58.80}$ | $-32.44_{\pm 66.26}$ | $-440.55_{\pm 437.28}$ | $-14.18_{\pm 22.04}$ | $-28.35_{\pm 44.07}$ | $-18.90_{\pm 29.38}$ | $-17.06_{\pm 26.51}$ | $-507.18_{\pm 469.96}$ | $-9.33_{\pm 14.61}$ | $-18.66_{\pm 29.21}$ | $-12.44_{\pm 19.48}$ | $-9.36_{\pm 14.66}$ |
| CMAES | $-57.83_{\pm 25.97}$ | $0.97_{\pm 0.00}$ | $1.95_{\pm 0.01}$ | $1.30_{\pm 0.01}$ | $1.46_{\pm 0.01}$ | - | - | - | - | - | - | - | - | - | - |
| TTDDEA | $-9.05e4_{\pm 2.18e5}$ | $-29.05_{\pm 61.40}$ | $-58.10_{\pm 122.80}$ | $-38.74_{\pm 81.87}$ | $-43.65_{\pm 92.25}$ | $-1.63e6_{\pm 4.22e6}$ | $-741.15_{\pm 1.92e3}$ | $-1.48e3_{\pm 3.83e3}$ | $-988.21_{\pm 2.56e3}$ | $-891.75_{\pm 2.30e3}$ | $-1.25e6_{\pm 2.97e6}$ | $-1.74e3_{\pm 3.51e3}$ | $-3.48e3_{\pm 7.02e3}$ | $-2.32e3_{\pm 4.68e3}$ | $-1.75e3_{\pm 3.52e3}$ |
| Trimentoring | $-41.93_{\pm 4.60}$ | $0.98_{\pm 0.00}$ | $1.96_{\pm 0.02}$ | $1.31_{\pm 0.01}$ | $1.48_{\pm 0.01}$ | $-41.89_{\pm 10.22}$ | $0.99_{\pm 0.01}$ | $1.97_{\pm 0.01}$ | $1.31_{\pm 0.01}$ | $1.19_{\pm 0.01}$ | $-42.02_{\pm 10.41}$ | $0.99_{\pm 0.01}$ | $1.97_{\pm 0.01}$ | $1.31_{\pm 0.01}$ | $0.99_{\pm 0.01}$ |
| Task | | | | | | | | GTOPX 4 | | | | | | | |
| $f(x^*_{\text{OFF}})$ | | | | | | | | $496.56_{\pm 5.01}$ | | | | | | | |
| ARCOO | $-130.02_{\pm 37.41}$ | $0.93_{\pm 0.02}$ | $1.86_{\pm 0.04}$ | $1.24_{\pm 0.03}$ | $1.40_{\pm 0.03}$ | $-147.14_{\pm 65.26}$ | $0.94_{\pm 0.02}$ | $1.87_{\pm 0.04}$ | $1.25_{\pm 0.03}$ | $1.13_{\pm 0.02}$ | $-144.18_{\pm 22.15}$ | $0.93_{\pm 0.03}$ | $1.87_{\pm 0.05}$ | $1.24_{\pm 0.03}$ | $0.94_{\pm 0.03}$ |
| BO | $-123.06_{\pm 19.08}$ | $0.95_{\pm 0.01}$ | $1.87_{\pm 0.02}$ | $1.26_{\pm 0.01}$ | $1.41_{\pm 0.01}$ | $-104.96_{\pm 11.74}$ | $0.94_{\pm 0.01}$ | $1.87_{\pm 0.02}$ | $1.25_{\pm 0.01}$ | $1.13_{\pm 0.01}$ | $-109.94_{\pm 17.67}$ | $0.94_{\pm 0.01}$ | $1.87_{\pm 0.02}$ | $1.25_{\pm 0.01}$ | $0.94_{\pm 0.01}$ |
| CBAS | $496.56_{\pm 5.01}$ | $1.00_{\pm 0.00}$ | $1.00_{\pm 0.00}$ | $1.00_{\pm 0.00}$ | $1.00_{\pm 0.00}$ | $496.56_{\pm 5.01}$ | $1.00_{\pm 0.00}$ | $1.00_{\pm 0.00}$ | $1.00_{\pm 0.00}$ | $1.00_{\pm 0.00}$ | $496.56_{\pm 5.01}$ | $1.00_{\pm 0.00}$ | $1.00_{\pm 0.00}$ | $1.00_{\pm 0.00}$ | $1.00_{\pm 0.00}$ |
| CCDDEA | $-8.19e3_{\pm 1.78e4}$ | $-376.50_{\pm 671.21}$ | $-753.01_{\pm 1.34e3}$ | $-502.01_{\pm 894.95}$ | $-565.69_{\pm 1.01e3}$ | $-1.11e7_{\pm 2.24e7}$ | $-1.05e3_{\pm 1.95e3}$ | $-2.09e3_{\pm 3.91e3}$ | $-1.40e3_{\pm 2.60e3}$ | $-1.26e3_{\pm 2.35e3}$ | $-2.76e6_{\pm 6.81e6}$ | $-2.00e3_{\pm 4.18e3}$ | $-3.99e3_{\pm 8.35e3}$ | $-2.66e3_{\pm 5.57e3}$ | $-2.00e3_{\pm 4.19e3}$ |
| CMAES | $-109.97_{\pm 28.76}$ | $0.95_{\pm 0.01}$ | $1.90_{\pm 0.01}$ | $1.27_{\pm 0.01}$ | $1.43_{\pm 0.01}$ | - | - | - | - | - | - | - | - | - | - |
| TTDDEA | $-1.78e4_{\pm 3.42}$ | $-19.04_{\pm 44.01}$ | $-38.08_{\pm 88.82}$ | $-25.38_{\pm 59.88}$ | $-28.61_{\pm 67.48}$ | $-7.42e6_{\pm 1.95e7}$ | $-845.49_{\pm 2.21e3}$ | $-1.69e3_{\pm 4.43}$ | $-1.13e3_{\pm 2.95e3}$ | $-1.02e3_{\pm 2.66e3}$ | $-5.39e7_{\pm 1.42e8}$ | $-1.07e4_{\pm 2.84e4}$ | $-2.15e4_{\pm 5.63e4}$ | $-1.43e4_{\pm 3.78e4}$ | $-1.08e4_{\pm 2.84e4}$ |
| Trimentoring | $-116.93_{\pm 19.65}$ | $0.94_{\pm 0.00}$ | $1.87_{\pm 0.01}$ | $1.25_{\pm 0.02}$ | $1.41_{\pm 0.00}$ | $-126.56_{\pm 21.71}$ | $0.94_{\pm 0.02}$ | $1.87_{\pm 0.01}$ | $1.25_{\pm 0.02}$ | $1.13_{\pm 0.02}$ | $-120.75_{\pm 26.25}$ | $0.94_{\pm 0.02}$ | $1.88_{\pm 0.03}$ | $1.25_{\pm 0.02}$ | $0.94_{\pm 0.02}$ |
| Task | | | | | | | | GTOPX 6 | | | | | | | |
| $f(x^*_{\text{OFF}})$ | | | | | | | | $-168.89_{\pm 0.62}$ | | | | | | | |
| ARCOO | $-51.98_{\pm 6.03}$ | $0.94_{\pm 0.01}$ | $1.87_{\pm 0.00}$ | $1.25_{\pm 0.00}$ | $1.41_{\pm 0.00}$ | $-56.44_{\pm 13.07}$ | $0.94_{\pm 0.01}$ | $1.88_{\pm 0.03}$ | $1.25_{\pm 0.02}$ | $1.13_{\pm 0.02}$ | $-57.11_{\pm 15.68}$ | $0.94_{\pm 0.01}$ | $1.87_{\pm 0.00}$ | $1.25_{\pm 0.04}$ | $0.94_{\pm 0.01}$ |
| BO | $-60.65_{\pm 11.30}$ | $0.90_{\pm 0.00}$ | $1.78_{\pm 0.02}$ | $1.20_{\pm 0.01}$ | $1.35_{\pm 0.00}$ | $-60.37_{\pm 11.18}$ | $0.90_{\pm 0.01}$ | $1.78_{\pm 0.02}$ | $1.20_{\pm 0.01}$ | $1.08_{\pm 0.01}$ | $-56.30_{\pm 4.91}$ | $0.89_{\pm 0.01}$ | $1.79_{\pm 0.02}$ | $1.19_{\pm 0.02}$ | $0.90_{\pm 0.01}$ |
| CBAS | $-168.89_{\pm 0.62}$ | $1.00_{\pm 0.00}$ | $1.00_{\pm 0.00}$ | $1.00_{\pm 0.00}$ | $1.00_{\pm 0.00}$ | $-168.89_{\pm 0.62}$ | $1.00_{\pm 0.00}$ | $1.00_{\pm 0.00}$ | $1.00_{\pm 0.00}$ | $1.00_{\pm 0.00}$ | $-168.89_{\pm 0.62}$ | $1.00_{\pm 0.00}$ | $1.00_{\pm 0.00}$ | $1.00_{\pm 0.00}$ | $1.00_{\pm 0.00}$ |
| CCDDEA | $-173.35_{\pm 106.97}$ | $0.56_{\pm 0.21}$ | $1.11_{\pm 0.42}$ | $0.74_{\pm 0.28}$ | $0.84_{\pm 0.32}$ | $-579.63_{\pm 480.01}$ | $0.22_{\pm 0.44}$ | $0.45_{\pm 0.88}$ | $0.30_{\pm 0.59}$ | $0.27_{\pm 0.53}$ | $-579.63_{\pm 480.01}$ | $-0.24_{\pm 0.94}$ | $-0.49_{\pm 1.89}$ | $-0.33_{\pm 1.26}$ | $-0.24_{\pm 0.95}$ |
| CMAES | $-44.05_{\pm 6.79}$ | $0.95_{\pm 0.01}$ | $1.91_{\pm 0.02}$ | $1.27_{\pm 0.01}$ | $1.43_{\pm 0.02}$ | - | - | - | - | - | - | - | - | - | - |
| TTDDEA | $-251.87_{\pm 333.39}$ | $0.39_{\pm 0.96}$ | $0.79_{\pm 1.92}$ | $0.52_{\pm 1.28}$ | $0.59_{\pm 1.44}$ | $-197.60_{\pm 218.58}$ | $0.14_{\pm 1.17}$ | $0.28_{\pm 2.34}$ | $0.19_{\pm 1.56}$ | $0.17_{\pm 1.41}$ | $-1.84e3_{\pm 4.55e3}$ | $-0.54_{\pm 2.76}$ | $-1.07_{\pm 5.51}$ | $-0.71_{\pm 3.68}$ | $-0.54_{\pm 2.77}$ |
| Trimentoring | $-59.93_{\pm 42.21}$ | $0.95_{\pm 0.04}$ | $1.78_{\pm 0.31}$ | $1.23_{\pm 0.10}$ | $1.37_{\pm 0.15}$ | $-65.96_{\pm 41.05}$ | $0.95_{\pm 0.04}$ | $1.78_{\pm 0.30}$ | $1.23_{\pm 0.10}$ | $1.12_{\pm 0.07}$ | $-66.16_{\pm 40.96}$ | $0.95_{\pm 0.05}$ | $1.77_{\pm 0.30}$ | $1.22_{\pm 0.10}$ | $0.95_{\pm 0.05}$ |

Table 6: Overall results for GTOPX unconstrained tasks with 128 solutions and 100th percentile evaluations. In this case, 0% of the values are missing near the worst value and another 40% near the optimal value. Details are the same as Table 5.

| Steps | t = 50 | | | | | t = 100 | | | | | t = 150 | | | | |
|---|---|---|---|---|---|---|---|---|---|---|---|---|---|---|---|
| Task $f(x^*_{OFF})$ | GTOPX 2 −275.27±2.54 | | | | | | | | | | | | | | |
| Metric | FS | SI | OI | SO | SO$_w$ | FS | SI | OI | SO | SO$_w$ | FS | SI | OI | SO | SO$_w$ |
| ARCOO | −63.93±6.87 | 0.97±0.01 | 1.93±0.02 | 1.29±0.02 | 1.45±0.02 | −63.14±6.40 | 0.98±0.01 | 1.95±0.03 | 1.30±0.02 | 1.17±0.02 | −63.23±6.58 | 0.98±0.01 | 1.96±0.03 | 1.31±0.02 | 0.98±0.01 |
| BO | −103.84±11.84 | 0.90±0.01 | 1.79±0.02 | 1.20±0.01 | 1.35±0.01 | −102.39±16.83 | 0.90±0.01 | 1.79±0.01 | 1.20±0.01 | 1.08±0.01 | −95.32±24.64 | 0.90±0.01 | 1.79±0.01 | 1.19±0.01 | 0.90±0.01 |
| CBAS | −275.27±2.34 | 1.00±0.00 | 1.00±0.00 | 1.00±0.00 | 1.00±0.00 | −275.27±2.34 | 1.00±0.00 | 1.00±0.00 | 1.00±0.00 | 1.00±0.00 | −275.27±2.34 | 1.00±0.00 | 1.00±0.00 | 1.00±0.00 | 1.00±0.00 |
| CCDDEA | −204.31±113.26 | 0.58±0.18 | 1.14±0.36 | 0.77±0.24 | 0.86±0.27 | −1.17e3±1.27e3 | −1.05±3.52 | −2.10±7.03 | −1.40±4.69 | −1.26±4.23 | −3.93e3±3.34e3 | −2.90±7.79 | −5.79±15.58 | −3.86±10.39 | −2.91±7.81 |
| CMAES | −58.11±4.25 | 0.98±0.01 | 1.95±0.02 | 1.30±0.01 | 1.46±0.02 | - | - | - | - | - | - | - | - | - | - |
| TTDDEA | −300.71±186.24 | 0.62±0.22 | 1.23±0.44 | 0.82±0.29 | 0.93±0.33 | −756.30±1.16e3 | 0.43±0.50 | 0.87±1.01 | 0.58±0.67 | 0.52±0.61 | −416.96±446.84 | 0.15±1.02 | 0.31±2.03 | 0.21±1.36 | 0.16±1.02 |
| Trimentoring | −59.43±7.51 | 0.98±0.01 | 1.95±0.02 | 1.30±0.01 | 1.47±0.01 | −57.60±8.28 | 0.98±0.01 | 1.96±0.02 | 1.31±0.01 | 1.18±0.01 | −58.80±9.35 | 0.98±0.01 | 1.97±0.01 | 1.31±0.01 | 0.99±0.01 |
| Task $f(x^*_{OFF})$ | GTOPX 3 −228.84±3.46 | | | | | | | | | | | | | | |
| ARCOO | −64.57±37.53 | 0.94±0.04 | 1.88±0.08 | 1.26±0.06 | 1.42±0.06 | −64.00±42.61 | 0.94±0.07 | 1.88±0.14 | 1.25±0.10 | 1.13±0.09 | −68.32±41.30 | 0.94±0.08 | 1.87±0.16 | 1.25±0.11 | 0.94±0.08 |
| BO | −71.88±18.09 | 0.92±0.01 | 1.81±0.01 | 1.22±0.01 | 1.37±0.01 | −68.57±13.93 | 0.91±0.00 | 1.82±0.01 | 1.22±0.01 | 1.10±0.01 | −68.51±9.46 | 0.91±0.01 | 1.82±0.01 | 1.21±0.01 | 0.91±0.01 |
| CBAS | −228.84±3.46 | 1.00±0.00 | 1.00±0.00 | 1.00±0.00 | 1.00±0.00 | −228.84±3.46 | 1.00±0.00 | 1.00±0.00 | 1.00±0.00 | 1.00±0.00 | −228.84±3.46 | 1.00±0.00 | 1.00±0.00 | 1.00±0.00 | 1.00±0.00 |
| CCDDEA | −49.670±10.826 | 0.96±0.02 | 1.92±0.04 | 1.28±0.02 | 1.45±0.03 | −293.53±339.51 | −5.63±11.73 | −11.26±23.46 | −7.50±15.64 | −6.77±14.12 | −294.16±339.29 | −3.63±8.09 | −7.26±16.18 | −4.84±10.79 | −3.64±8.12 |
| CMAES | −1.38e6±3.55e6 | −1.19e3±3.01e3 | −2.37e3±6.01e3 | −1.58e3±4.01e3 | −1.78e3±4.52e3 | −5.56e5±1.10e6 | −1.82e3±4.55e3 | −3.65e3±9.10e3 | −2.43e3±6.06e3 | −2.19e3±5.47e3 | −2.37e6±4.11e6 | −9.37e3±1.83e4 | −1.87e4±3.67e4 | −1.25e4±2.45e4 | −9.40e3±1.84e4 |
| Trimentoring | −39.57±7.43 | 0.97±0.01 | 1.95±0.03 | 1.30±0.02 | 1.46±0.02 | −40.80±4.58 | 0.98±0.01 | 1.96±0.02 | 1.30±0.02 | 1.18±0.01 | −45.12±4.43 | 0.98±0.01 | 1.96±0.02 | 1.30±0.01 | 0.98±0.01 |
| Task $f(x^*_{OFF})$ | GTOPX 4 −322.50±1.08 | | | | | | | | | | | | | | |
| ARCOO | −156.34±64.20 | 0.89±0.11 | 1.75±0.22 | 1.18±0.15 | 1.32±0.17 | −171.14±57.54 | 0.86±0.12 | 1.69±0.23 | 1.14±0.16 | 1.03±0.14 | −173.20±76.27 | 0.84±0.11 | 1.68±0.23 | 1.12±0.15 | 0.84±0.11 |
| BO | −115.12±26.78 | 0.91±0.01 | 1.80±0.03 | 1.21±0.01 | 1.36±0.01 | −105.98±14.88 | 0.90±0.01 | 1.80±0.02 | 1.20±0.01 | 1.09±0.01 | −111.40±13.69 | 0.90±0.01 | 1.80±0.02 | 1.20±0.01 | 0.90±0.01 |
| CBAS | −322.50±1.08 | 1.00±0.00 | 1.00±0.00 | 1.00±0.00 | 1.00±0.00 | −322.50±1.08 | 1.00±0.00 | 1.00±0.00 | 1.00±0.00 | 1.00±0.00 | −322.50±1.08 | 1.00±0.00 | 1.00±0.00 | 1.00±0.00 | 1.00±0.00 |
| CCDDEA | −119.20±39.68 | 0.93±0.02 | 1.86±0.03 | 1.24±0.02 | 1.40±0.02 | −2.14e5±1.89e5 | −3.74e3±6.06e3 | −7.49e3±1.61e4 | −4.99e3±1.07e4 | −4.50e3±9.69e3 | −3.41e5±4.06e5 | −2.72e3±3.44e3 | −5.44e3±1.09e4 | −3.62e3±7.26e3 | −2.72e3±5.46e3 |
| CMAES | −177.08±401.38 | 0.17±0.60 | −354.17±802.76 | 1.86±0.03 | 1.24±0.02 | −266.07±603.07 | - | - | - | - | −195.41±109.56 | - | - | - | - |
| TTDDEA | −4.41e3±1.00e4 | −4.42±11.68 | −8.83±23.37 | −5.89±15.58 | −6.64±17.59 | −2.43e5±6.40e5 | −10.00±24.26 | −19.99±48.57 | −13.33±32.38 | −12.03±29.22 | −4.68e4±1.12e5 | −44.89±115.20 | −89.77±230.40 | −59.85±153.60 | −45.04±115.58 |
| Trimentoring | −116.56±32.40 | 0.89±0.06 | 1.77±0.12 | 1.19±0.07 | 1.33±0.09 | −140.96±110.67 | 0.87±0.11 | 1.73±0.22 | 1.16±0.15 | 1.04±0.13 | −168.45±170.22 | 0.84±0.17 | 1.68±0.33 | 1.12±0.22 | 0.84±0.17 |
| Task $f(x^*_{OFF})$ | GTOPX 6 −142.09±0.39 | | | | | | | | | | | | | | |
| ARCOO | −47.28±6.35 | 0.94±0.01 | 1.89±0.02 | 1.26±0.01 | 1.42±0.01 | −49.08±9.14 | 0.95±0.02 | 1.90±0.04 | 1.26±0.02 | 1.14±0.02 | −49.30±9.46 | 0.95±0.03 | 1.89±0.05 | 1.26±0.03 | 0.95±0.03 |
| BO | −59.47±13.67 | 0.87±0.01 | 1.73±0.03 | 1.16±0.02 | 1.30±0.02 | −62.86±7.96 | 0.87±0.01 | 1.73±0.02 | 1.16±0.02 | 1.04±0.01 | −53.66±19.84 | 0.86±0.01 | 1.73±0.02 | 1.15±0.01 | 0.87±0.00 |
| CBAS | −142.09±0.39 | 1.00±0.00 | 1.00±0.00 | 1.00±0.00 | 1.00±0.00 | −142.09±0.39 | 1.00±0.00 | 1.00±0.00 | 1.00±0.00 | 1.00±0.00 | −142.09±0.39 | 1.00±0.00 | 1.00±0.00 | 1.00±0.00 | 1.00±0.00 |
| CCDDEA | −203.18±115.84 | 0.19±0.37 | 0.38±0.74 | 0.25±0.49 | 0.29±0.95 | −256.00±191.92 | 0.04±0.54 | 0.07±1.09 | 0.05±0.72 | 0.04±0.65 | −195.41±109.96 | 0.06±0.52 | 0.13±1.04 | 0.08±0.70 | 0.06±0.52 |
| CMAES | −41.53±5.42 | 0.94±0.02 | 1.89±0.05 | 1.26±0.02 | 1.42±0.03 | - | - | - | - | - | - | - | - | - | - |
| TTDDEA | −105.50±28.72 | 0.74±0.07 | 1.48±0.15 | 0.99±0.10 | 1.11±0.11 | −104.90±25.79 | 0.71±0.09 | 1.42±0.19 | 0.95±0.12 | 0.86±0.11 | −110.22±25.60 | 0.70±0.10 | 1.40±0.19 | 0.93±0.13 | 0.70±0.10 |
| Trimentoring | −65.81±30.08 | 0.94±0.03 | 1.74±0.28 | 1.21±0.08 | 1.34±0.13 | −65.55±30.69 | 0.93±0.04 | 1.72±0.27 | 1.19±0.08 | 1.09±0.04 | −66.56±30.34 | 0.92±0.04 | 1.72±0.27 | 1.19±0.08 | 0.92±0.04 |

Table 7: Overall results for GTOPX unconstrained tasks with 128 solutions and 100th percentile evaluations. In this case, 0% of the values are missing near the worst value and another 30% near the optimal value. Details are the same as Table 5.

| Steps | t = 50 | | | | | t = 100 | | | | | t = 150 | | | | |
|---|---|---|---|---|---|---|---|---|---|---|---|---|---|---|---|
| Task $f(x^*_{OFF})$ | GTOPX 2 −218.12±1.78 | | | | | | | | | | | | | | |
| Metric | FS | SI | OI | SO | SO$_w$ | FS | SI | OI | SO | SO$_w$ | FS | SI | OI | SO | SO$_w$ |
| ARCOO | −60.17±7.67 | 0.97±0.01 | 1.93±0.02 | 1.29±0.02 | 1.45±0.02 | −61.62±8.21 | 0.97±0.01 | 1.95±0.03 | 1.30±0.02 | 1.17±0.02 | −61.34±8.92 | 0.98±0.01 | 1.95±0.03 | 1.30±0.02 | 0.98±0.01 |
| BO | −95.21±15.38 | 0.88±0.01 | 1.73±0.02 | 1.16±0.01 | 1.31±0.01 | −107.04±13.78 | 0.87±0.01 | 1.73±0.02 | 1.16±0.01 | 1.05±0.01 | −97.91±18.57 | 0.87±0.01 | 1.73±0.01 | 1.16±0.01 | 0.87±0.00 |
| CBAS | −218.12±1.78 | 1.00±0.00 | 1.00±0.00 | 1.00±0.00 | 1.00±0.00 | −218.12±1.78 | 1.00±0.00 | 1.00±0.00 | 1.00±0.00 | 1.00±0.00 | −218.12±1.78 | 1.00±0.00 | 1.00±0.00 | 1.00±0.00 | 1.00±0.00 |
| CCDDEA | −288.63±280.89 | 0.17±0.60 | 0.33±1.20 | 0.22±0.80 | 0.25±0.90 | −3.10e2±2.77e3 | −3.07±3.76 | −6.14±7.52 | −4.09±5.01 | −3.69±4.52 | −7.55e3±1.36e4 | −9.99±17.85 | −19.99±35.70 | −13.32±23.80 | −10.03±17.91 |
| CMAES | −60.92±9.36 | 0.97±0.01 | 1.94±0.03 | 1.29±0.02 | 1.46±0.00 | - | - | - | - | - | - | - | - | - | - |
| TTDDEA | −223.21±81.03 | 0.56±0.09 | 1.12±0.17 | 0.74±0.12 | 0.84±0.13 | −243.58±59.05 | 0.49±0.15 | 0.98±0.29 | 0.65±0.20 | 0.56±0.58 | −245.93±60.48 | 0.47±0.16 | 0.93±0.33 | 0.62±0.22 | 0.47±0.16 |
| Trimentoring | −78.67±35.85 | 0.98±0.01 | 1.83±0.32 | 1.27±0.10 | 1.41±0.18 | −78.93±33.64 | 0.99±0.01 | 1.85±0.32 | 1.27±0.10 | 1.16±0.06 | −81.95±52.36 | 0.99±0.01 | 1.83±0.32 | 1.27±0.10 | 0.99±0.01 |
| Task $f(x^*_{OFF})$ | GTOPX 3 −171.54±0.81 | | | | | | | | | | | | | | |
| ARCOO | −51.89±9.89 | 0.96±0.04 | 1.89±0.05 | 1.27±0.02 | 1.43±0.03 | −47.77±9.88 | 0.96±0.02 | 1.90±0.06 | 1.27±0.03 | 1.15±0.02 | −44.14±9.60 | 0.96±0.02 | 1.91±0.08 | 1.28±0.03 | 0.96±0.02 |
| BO | −67.29±9.47 | 0.88±0.01 | 1.74±0.02 | 1.17±0.01 | 1.31±0.02 | −77.44±14.99 | 0.88±0.01 | 1.74±0.02 | 1.17±0.01 | 1.06±0.01 | −67.66±23.97 | 0.87±0.01 | 1.74±0.02 | 1.16±0.01 | 0.87±0.00 |
| CBAS | −171.54±0.81 | 1.00±0.00 | 1.00±0.00 | 1.00±0.00 | 1.00±0.00 | −171.54±0.81 | 1.00±0.00 | 1.00±0.00 | 1.00±0.00 | 1.00±0.00 | −171.54±0.81 | 1.00±0.00 | 1.00±0.00 | 1.00±0.00 | 1.00±0.00 |
| CCDDEA | −3.89e3±4.97e3 | −21.45±28.04 | −42.84±56.12 | −28.59±37.40 | −32.20±42.15 | −679.88±1.07e3 | −16.79±17.26 | −33.58±34.52 | −22.38±23.01 | −20.20±20.77 | −730.69±1083.44 | −11.68±21.21 | −23.36±22.42 | −15.57±14.94 | −11.72±11.25 |
| CMAES | −48.03±7.74 | 0.95±0.02 | 1.89±0.04 | 1.26±0.03 | 1.42±0.00 | - | - | - | - | - | - | - | - | - | - |
| TTDDEA | −3.01e5±4.87e5 | −754.51±1.25e3 | −1.51e3±2.50e3 | −1.01e3±1.66e3 | −1.13e3±1.87e3 | −6.59e7±1.75e8 | −9.36e3±2.36e4 | −1.87e4±4.71e4 | −1.25e4±3.14e4 | −1.13e4±2.83e4 | −3.31e6±6.42e6 | −8.20e4±2.09e5 | −1.64e5±4.19e5 | −1.09e5±2.79e5 | −8.23e4±2.10e5 |
| Trimentoring | −44.05±6.23 | 0.97±0.01 | 1.92±0.03 | 1.29±0.01 | 1.45±0.02 | −41.33±8.59 | 0.97±0.01 | 1.93±0.03 | 1.29±0.02 | 1.17±0.01 | −40.80±10.22 | 0.97±0.01 | 1.94±0.03 | 1.29±0.02 | 0.97±0.01 |
| Task $f(x^*_{OFF})$ | GTOPX 4 −241.61±1.39 | | | | | | | | | | | | | | |
| ARCOO | −134.37±21.45 | 0.83±0.03 | 1.66±0.06 | 1.11±0.04 | 1.25±0.05 | −143.98±49.97 | 0.82±0.05 | 1.63±0.09 | 1.09±0.06 | 0.98±0.05 | −165.41±49.65 | 0.80±0.08 | 1.61±0.15 | 1.07±0.10 | 0.81±0.08 |
| BO | −111.22±26.29 | 0.87±0.01 | 1.71±0.02 | 1.16±0.01 | 1.30±0.01 | −103.39±13.47 | 0.86±0.01 | 1.72±0.03 | 1.15±0.02 | 1.04±0.02 | −113.88±21.70 | 0.86±0.01 | 1.73±0.01 | 1.16±0.01 | 0.87±0.00 |
| CBAS | −241.61±1.39 | 1.00±0.00 | 1.00±0.00 | 1.00±0.00 | 1.00±0.00 | −241.61±1.39 | 1.00±0.00 | 1.00±0.00 | 1.00±0.00 | 1.00±0.00 | −241.61±1.39 | 1.00±0.00 | 1.00±0.00 | 1.00±0.00 | 1.00±0.00 |
| CCDDEA | −3.32e3±2.55e3 | −1.67e3±3.87e3 | −3.34e3±7.73e3 | −2.23e3±5.15e3 | −2.51e3±5.81e3 | −2.42e5±2.37e5 | −1.11e3±1.95e3 | −2.23e3±3.90e3 | −1.48e3±2.60e3 | −1.34e3±2.34e3 | −5.79e5±9.88e5 | −1.18e4±2.61e4 | −2.35e4±5.23e4 | −1.57e4±3.49e4 | −1.18e4±2.62e4 |
| CMAES | −124.10±81.74 | 0.90±0.03 | 1.80±0.06 | 1.20±0.04 | 1.35±0.04 | - | - | - | - | - | - | - | - | - | - |
| TTDDEA | −196.12±101.00 | 0.76±0.12 | 1.50±0.21 | 1.01±0.15 | 1.13±0.17 | −674.13±1144.16 | 0.47±0.49 | 0.94±0.97 | 0.62±0.65 | 0.56±0.58 | −643.53±1090.20 | −0.04±1.67 | −0.08±3.34 | −0.06±2.23 | −0.04±1.68 |
| Trimentoring | −123.00±21.02 | 0.84±0.04 | 1.68±0.09 | 1.12±0.06 | 1.26±0.07 | −126.66±33.24 | 0.83±0.04 | 1.66±0.09 | 1.11±0.06 | 1.00±0.05 | −115.20±32.26 | 0.84±0.05 | 1.68±0.10 | 1.12±0.07 | 0.84±0.05 |
| Task $f(x^*_{OFF})$ | GTOPX 6 −121.49±0.38 | | | | | | | | | | | | | | |
| ARCOO | −55.11±14.08 | 0.92±0.04 | 1.85±0.05 | 1.23±0.04 | 1.39±0.04 | −54.21±12.66 | 0.91±0.04 | 1.83±0.08 | 1.22±0.06 | 1.10±0.05 | −56.26±14.96 | 0.91±0.05 | 1.81±0.09 | 1.21±0.06 | 0.91±0.05 |
| BO | −65.88±10.55 | 0.86±0.04 | 1.67±0.03 | 1.14±0.02 | 1.27±0.02 | −56.08±13.93 | 0.85±0.02 | 1.68±0.04 | 1.13±0.03 | 1.02±0.03 | −61.06±15.55 | 0.84±0.02 | 1.68±0.04 | 1.12±0.03 | 0.84±0.02 |
| CBAS | −121.48±0.38 | 1.00±0.00 | 1.00±0.00 | 1.00±0.00 | 1.00±0.00 | −121.48±0.38 | 1.00±0.00 | 1.00±0.00 | 1.00±0.00 | 1.00±0.00 | −121.48±0.38 | 1.00±0.00 | 1.00±0.00 | 1.00±0.00 | 1.00±0.00 |
| CCDDEA | −108.83±35.33 | 0.15±0.64 | 0.31±1.22 | 0.20±0.84 | 0.23±0.95 | −202.87±123.85 | 0.23±0.33 | 0.45±0.67 | 0.30±0.45 | 0.27±0.40 | −208.17±109.56 | 0.14±0.38 | 0.28±0.76 | 0.19±0.51 | 0.14±0.38 |
| CMAES | −45.24±11.31 | 0.94±0.04 | 1.87±0.07 | 1.25±0.05 | 1.41±0.05 | - | - | - | - | - | - | - | - | - | - |
| TTDDEA | −225.02±245.93 | −0.22±2.32 | −0.45±4.65 | −0.30±3.10 | −0.34±3.49 | −560.49±1185.23 | −0.13±1.57 | −0.25±3.14 | −0.17±2.09 | −0.15±1.89 | −955.96±3512.86 | −1.34±3.61 | −2.68±7.22 | −1.79±4.82 | −1.35±3.62 |
| Trimentoring | −53.56±25.57 | 0.96±0.04 | 1.92±0.05 | 1.28±0.03 | 1.44±0.04 | −55.68±25.09 | 0.95±0.03 | 1.90±0.06 | 1.27±0.04 | 1.15±0.03 | −57.89±24.13 | 0.94±0.04 | 1.89±0.06 | 1.26±0.04 | 0.95±0.03 |

Table 8: Overall results for GTOPX unconstrained tasks with 128 solutions and 100th percentile evaluations. In this case, 0% of the values are missing near the worst value and another 20% near the optimal value. Details are the same as Table 5.

| Steps | | | t = 50 | | | | | t = 100 | | | | | t = 150 | | |
|---|---|---|---|---|---|---|---|---|---|---|---|---|---|---|---|
| Task | | | | | | | | GTOPX 2 | | | | | | | |
| $f(x^*_{\mathrm{OFF}})$ | | | | | | | | $-175.10_{\pm 0.75}$ | | | | | | | |
| Metric | FS | SI | OI | SO | SO$_\omega$ | FS | SI | OI | SO | SO$_\omega$ | FS | SI | OI | SO | SO$_\omega$ |
| ARCOO | $-62.22_{\pm 5.92}$ | $0.96_{\pm 0.01}$ | $1.91_{\pm 0.02}$ | $1.28_{\pm 0.02}$ | $1.44_{\pm 0.02}$ | $-61.73_{\pm 4.92}$ | $0.97_{\pm 0.01}$ | $1.93_{\pm 0.07}$ | $1.29_{\pm 0.02}$ | $1.16_{\pm 0.02}$ | $-62.05_{\pm 6.28}$ | $0.97_{\pm 0.01}$ | $1.94_{\pm 0.03}$ | $1.29_{\pm 0.02}$ | $0.97_{\pm 0.01}$ |
| BO | $-103.86_{\pm 9.37}$ | $0.83_{\pm 0.02}$ | $1.64_{\pm 0.04}$ | $1.10_{\pm 0.03}$ | $1.24_{\pm 0.03}$ | $-104.05_{\pm 24.97}$ | $0.83_{\pm 0.02}$ | $1.64_{\pm 0.03}$ | $1.10_{\pm 0.02}$ | $0.99_{\pm 0.02}$ | $-93.98_{\pm 13.90}$ | $0.82_{\pm 0.01}$ | $1.64_{\pm 0.03}$ | $1.09_{\pm 0.02}$ | $0.82_{\pm 0.01}$ |
| CBAS | $-175.10_{\pm 0.75}$ | $1.00_{\pm 0.00}$ | $1.00_{\pm 0.00}$ | $1.00_{\pm 0.00}$ | $1.00_{\pm 0.00}$ | $-175.10_{\pm 0.75}$ | $1.00_{\pm 0.00}$ | $1.00_{\pm 0.00}$ | $1.00_{\pm 0.00}$ | $1.00_{\pm 0.00}$ | $-175.10_{\pm 0.75}$ | $1.00_{\pm 0.00}$ | $1.00_{\pm 0.00}$ | $1.00_{\pm 0.00}$ | $1.00_{\pm 0.00}$ |
| CCDDEA | $-238.92_{\pm 218.49}$ | $0.41_{\pm 0.21}$ | $0.81_{\pm 0.42}$ | $0.54_{\pm 0.28}$ | $0.61_{\pm 0.31}$ | $-1.98e3_{\pm 3.03e5}$ | $-0.96_{\pm 2.28}$ | $-1.93_{\pm 4.55}$ | $-1.28_{\pm 3.04}$ | $-1.16_{\pm 2.74}$ | $-2.04e3_{\pm 3.00e3}$ | $-3.30_{\pm 6.08}$ | $-6.61_{\pm 12.15}$ | $-4.40_{\pm 8.10}$ | $-3.31_{\pm 6.10}$ |
| CMAES | $-60.46_{\pm 7.27}$ | $0.96_{\pm 0.02}$ | $1.92_{\pm 0.04}$ | $1.28_{\pm 0.03}$ | $1.45_{\pm 0.03}$ | - | - | - | - | - | - | - | - | - | - |
| TTDDEA | $-1.06e3_{\pm 2.12e3}$ | $-2.22_{\pm 6.69}$ | $-4.44_{\pm 13.38}$ | $-2.96_{\pm 8.92}$ | $-3.34_{\pm 10.05}$ | $-2457.03_{\pm 5845.33}$ | $-4.22_{\pm 11.81}$ | $-8.43_{\pm 23.62}$ | $-5.62_{\pm 15.74}$ | $-5.07_{\pm 14.21}$ | $-3020.72_{\pm 7250.61}$ | $-5.89_{\pm 16.12}$ | $-11.78_{\pm 32.23}$ | $-7.85_{\pm 21.49}$ | $-5.91_{\pm 16.17}$ |
| Trimentoring | $-57.88_{\pm 5.96}$ | $0.97_{\pm 0.01}$ | $1.93_{\pm 0.02}$ | $1.29_{\pm 0.01}$ | $1.46_{\pm 0.01}$ | $-58.90_{\pm 6.03}$ | $0.97_{\pm 0.01}$ | $1.94_{\pm 0.03}$ | $1.30_{\pm 0.01}$ | $1.17_{\pm 0.01}$ | $-57.67_{\pm 4.22}$ | $0.97_{\pm 0.01}$ | $1.94_{\pm 0.02}$ | $1.29_{\pm 0.02}$ | $0.97_{\pm 0.01}$ |
| Task | | | | | | | | GTOPX 3 | | | | | | | |
| $f(x^*_{\mathrm{OFF}})$ | | | | | | | | $-134.83_{\pm 0.56}$ | | | | | | | |
| ARCOO | $-46.56_{\pm 7.23}$ | $0.95_{\pm 0.02}$ | $1.89_{\pm 0.05}$ | $1.26_{\pm 0.03}$ | $1.42_{\pm 0.04}$ | $-45.86_{\pm 7.29}$ | $0.95_{\pm 0.02}$ | $1.90_{\pm 0.04}$ | $1.27_{\pm 0.03}$ | $1.14_{\pm 0.02}$ | $-45.80_{\pm 8.44}$ | $0.95_{\pm 0.02}$ | $1.90_{\pm 0.04}$ | $1.27_{\pm 0.03}$ | $0.95_{\pm 0.02}$ |
| BO | $-61.23_{\pm 11.20}$ | $0.85_{\pm 0.02}$ | $1.66_{\pm 0.03}$ | $1.12_{\pm 0.03}$ | $1.26_{\pm 0.03}$ | $-70.31_{\pm 22.42}$ | $0.84_{\pm 0.02}$ | $1.66_{\pm 0.02}$ | $1.11_{\pm 0.02}$ | $1.01_{\pm 0.02}$ | $-77.95_{\pm 22.01}$ | $0.83_{\pm 0.01}$ | $1.66_{\pm 0.02}$ | $1.11_{\pm 0.01}$ | $0.83_{\pm 0.01}$ |
| CBAS | $-134.83_{\pm 0.56}$ | $1.00_{\pm 0.00}$ | $1.00_{\pm 0.00}$ | $1.00_{\pm 0.00}$ | $1.00_{\pm 0.00}$ | $-134.83_{\pm 0.56}$ | $1.00_{\pm 0.00}$ | $1.00_{\pm 0.00}$ | $1.00_{\pm 0.00}$ | $1.00_{\pm 0.00}$ | $-134.83_{\pm 0.56}$ | $1.00_{\pm 0.00}$ | $1.00_{\pm 0.00}$ | $1.00_{\pm 0.00}$ | $1.00_{\pm 0.00}$ |
| CCDDEA | $-4.68e5_{\pm 1.25e6}$ | $-188.10_{\pm 542.21}$ | $-375.96_{\pm 1,684.34}$ | $-250.74_{\pm 496.31}$ | $-282.52_{\pm 514.22}$ | $-603.90_{\pm 1128.59}$ | $-130.77_{\pm 229.75}$ | $-261.51_{\pm 489.51}$ | $-174.36_{\pm 306.33}$ | $-157.34_{\pm 276.43}$ | $-626.14_{\pm 1119.29}$ | $-87.63_{\pm 152.40}$ | $-175.26_{\pm 304.81}$ | $-116.84_{\pm 201.20}$ | $-87.92_{\pm 152.91}$ |
| CMAES | $-51.07_{\pm 10.48}$ | $0.94_{\pm 0.02}$ | $1.89_{\pm 0.04}$ | $1.26_{\pm 0.03}$ | $1.42_{\pm 0.03}$ | - | - | - | - | - | - | - | - | - | - |
| TTDDEA | $-3.48e6_{\pm 8.87e6}$ | $-2.62e4_{\pm 6.73e4?}$ | $-5.25e4_{\pm 134540.03}$ | $-3.50e4_{\pm 8.97e4}$ | $-3.94e4_{\pm 1.01e5}$ | $-3.14e6_{\pm 7.34e6}$ | $-3.67e4_{\pm 9.37e4}$ | $-7.35e4_{\pm 1.97e5}$ | $-4.90e4_{\pm 1.25e5}$ | $-4.42e4_{\pm 1.13e5}$ | $-9.09e6_{\pm 1.43e7}$ | $-2.52e5_{\pm 5.84e5}$ | $-5.05e5_{\pm 1.17e6}$ | $-3.37e5_{\pm 7.78e5}$ | $-2.53e5_{\pm 5.86e5}$ |
| Trimentoring | $-55.90_{\pm 30.39}$ | $0.96_{\pm 0.02}$ | $1.78_{\pm 0.28}$ | $1.24_{\pm 0.09}$ | $1.38_{\pm 0.14}$ | $-56.78_{\pm 29.79}$ | $0.96_{\pm 0.03}$ | $1.79_{\pm 0.29}$ | $1.24_{\pm 0.09}$ | $1.13_{\pm 0.05}$ | $-55.95_{\pm 29.96}$ | $0.96_{\pm 0.03}$ | $1.79_{\pm 0.29}$ | $1.24_{\pm 0.09}$ | $0.96_{\pm 0.02}$ |
| Task | | | | | | | | GTOPX 4 | | | | | | | |
| $f(x^*_{\mathrm{OFF}})$ | | | | | | | | $-194.03_{\pm 0.88}$ | | | | | | | |
| ARCOO | $-149.32_{\pm 37.43}$ | $0.75_{\pm 0.04}$ | $1.50_{\pm 0.09}$ | $1.00_{\pm 0.06}$ | $1.13_{\pm 0.07}$ | $-144.90_{\pm 24.33}$ | $0.72_{\pm 0.08}$ | $1.43_{\pm 0.17}$ | $0.95_{\pm 0.11}$ | $0.86_{\pm 0.10}$ | $-146.42_{\pm 36.69}$ | $0.71_{\pm 0.07}$ | $1.41_{\pm 0.15}$ | $0.94_{\pm 0.10}$ | $0.71_{\pm 0.07}$ |
| BO | $-107.50_{\pm 11.78}$ | $0.83_{\pm 0.02}$ | $1.60_{\pm 0.04}$ | $1.09_{\pm 0.03}$ | $1.22_{\pm 0.04}$ | $-111.09_{\pm 8.67}$ | $0.80_{\pm 0.01}$ | $1.60_{\pm 0.03}$ | $1.07_{\pm 0.02}$ | $0.96_{\pm 0.02}$ | $-120.46_{\pm 17.03}$ | $0.80_{\pm 0.01}$ | $1.60_{\pm 0.02}$ | $1.07_{\pm 0.02}$ | $0.80_{\pm 0.01}$ |
| CBAS | $-194.03_{\pm 0.88}$ | $1.00_{\pm 0.00}$ | $1.00_{\pm 0.00}$ | $1.00_{\pm 0.00}$ | $1.00_{\pm 0.00}$ | $-194.03_{\pm 0.88}$ | $1.00_{\pm 0.00}$ | $1.00_{\pm 0.00}$ | $1.00_{\pm 0.00}$ | $1.00_{\pm 0.00}$ | $-194.03_{\pm 0.88}$ | $1.00_{\pm 0.00}$ | $1.00_{\pm 0.00}$ | $1.00_{\pm 0.00}$ | $1.00_{\pm 0.00}$ |
| CCDDEA | $-1.45e6_{\pm 3.84e6}$ | $-388.94_{\pm 806.99}$ | $-777.87_{\pm 1.21e3}$ | $-518.58_{\pm 809.32}$ | $-584.37_{\pm 912.00}$ | $-1.50e6_{\pm 3.56e6}$ | $-2.00e3_{\pm 3.16e3}$ | $-4.01e3_{\pm 6.32e3}$ | $-2.67e3_{\pm 4.21e3}$ | $-2.41e3_{\pm 3.80e3}$ | $-2.99e6_{\pm 4.26e6}$ | $-5.81e3_{\pm 7.89e3}$ | $-1.16e4_{\pm 1.58e4}$ | $-7.75e3_{\pm 1.05e4}$ | $-5.83e3_{\pm 7.96e3}$ |
| CMAES | $-102.71_{\pm 20.57}$ | $0.88_{\pm 0.02}$ | $1.76_{\pm 0.05}$ | $1.17_{\pm 0.03}$ | $1.32_{\pm 0.03}$ | - | - | - | - | - | - | - | - | - | - |
| TTDDEA | $-9.11e3_{\pm 2.31e4}$ | $-27.61_{\pm 73.40}$ | $-55.21_{\pm 146.99}$ | $-36.81_{\pm 97.99}$ | $-41.48_{\pm 110.42}$ | $-1.67e4_{\pm 4.28e4}$ | $-95.51_{\pm 230.17}$ | $-191.01_{\pm 500.34}$ | $-127.34_{\pm 333.56}$ | $-114.91_{\pm 301.00}$ | $-9.68e3_{\pm 2.31e4}$ | $-79.81_{\pm 206.50}$ | $-159.62_{\pm 413.00}$ | $-106.41_{\pm 275.34}$ | $-80.07_{\pm 207.19}$ |
| Trimentoring | $-127.28_{\pm 23.35}$ | $0.82_{\pm 0.09}$ | $1.59_{\pm 0.14}$ | $1.08_{\pm 0.11}$ | $1.21_{\pm 0.12}$ | $-107.48_{\pm 37.66}$ | $0.81_{\pm 0.08}$ | $1.62_{\pm 0.16}$ | $1.08_{\pm 0.11}$ | $0.98_{\pm 0.10}$ | $-105.52_{\pm 19.98}$ | $0.82_{\pm 0.08}$ | $1.64_{\pm 0.17}$ | $1.09_{\pm 0.11}$ | $0.82_{\pm 0.08}$ |
| Task | | | | | | | | GTOPX 6 | | | | | | | |
| $f(x^*_{\mathrm{OFF}})$ | | | | | | | | $-102.96_{\pm 0.29}$ | | | | | | | |
| ARCOO | $-49.34_{\pm 7.29}$ | $0.89_{\pm 0.04}$ | $1.78_{\pm 0.09}$ | $1.19_{\pm 0.05}$ | $1.34_{\pm 0.04}$ | $-50.25_{\pm 5.39}$ | $0.88_{\pm 0.04}$ | $1.77_{\pm 0.07}$ | $1.18_{\pm 0.05}$ | $1.06_{\pm 0.04}$ | $-50.79_{\pm 9.21}$ | $0.88_{\pm 0.02}$ | $1.76_{\pm 0.08}$ | $1.17_{\pm 0.05}$ | $0.88_{\pm 0.04}$ |
| BO | $-68.35_{\pm 10.45}$ | $0.82_{\pm 0.01}$ | $1.59_{\pm 0.04}$ | $1.08_{\pm 0.02}$ | $1.21_{\pm 0.02}$ | $-62.75_{\pm 5.35}$ | $0.81_{\pm 0.02}$ | $1.59_{\pm 0.04}$ | $1.07_{\pm 0.03}$ | $0.97_{\pm 0.03}$ | $-66.49_{\pm 15.27}$ | $0.80_{\pm 0.02}$ | $1.59_{\pm 0.04}$ | $1.06_{\pm 0.03}$ | $0.80_{\pm 0.02}$ |
| CBAS | $-102.96_{\pm 0.29}$ | $1.00_{\pm 0.00}$ | $1.00_{\pm 0.00}$ | $1.00_{\pm 0.00}$ | $1.00_{\pm 0.00}$ | $-102.96_{\pm 0.29}$ | $1.00_{\pm 0.00}$ | $1.00_{\pm 0.00}$ | $1.00_{\pm 0.00}$ | $1.00_{\pm 0.00}$ | $-102.96_{\pm 0.29}$ | $1.00_{\pm 0.00}$ | $1.00_{\pm 0.00}$ | $1.00_{\pm 0.00}$ | $1.00_{\pm 0.00}$ |
| CCDDEA | $-93.76_{\pm 40.30}$ | $0.31_{\pm 0.18}$ | $0.59_{\pm 0.34}$ | $0.41_{\pm 0.23}$ | $0.46_{\pm 0.26}$ | $-6.79e3_{\pm 1.75e4}$ | $-14.23_{\pm 37.96}$ | $-28.46_{\pm 75.93}$ | $-18.97_{\pm 50.62}$ | $-17.12_{\pm 45.68}$ | $-6.79e3_{\pm 1.75e4}$ | $-29.64_{\pm 78.46}$ | $-59.29_{\pm 156.92}$ | $-39.52_{\pm 104.62}$ | $-29.74_{\pm 78.72}$ |
| CMAES | $-41.87_{\pm 6.32}$ | $0.91_{\pm 0.03}$ | $1.82_{\pm 0.07}$ | $1.22_{\pm 0.04}$ | $1.37_{\pm 0.05}$ | - | - | - | - | - | - | - | - | - | - |
| TTDDEA | $-110.60_{\pm 33.99}$ | $0.58_{\pm 0.15}$ | $1.16_{\pm 0.30}$ | $0.77_{\pm 0.20}$ | $0.87_{\pm 0.23}$ | $-119.60_{\pm 34.09}$ | $0.51_{\pm 0.14}$ | $1.02_{\pm 0.28}$ | $0.68_{\pm 0.18}$ | $0.62_{\pm 0.17}$ | $-120.60_{\pm 44.97}$ | $0.47_{\pm 0.17}$ | $0.93_{\pm 0.33}$ | $0.62_{\pm 0.22}$ | $0.47_{\pm 0.17}$ |
| Trimentoring | $-65.81_{\pm 29.31}$ | $0.94_{\pm 0.05}$ | $1.52_{\pm 0.42}$ | $1.14_{\pm 0.13}$ | $1.24_{\pm 0.20}$ | $-65.96_{\pm 29.80}$ | $0.94_{\pm 0.05}$ | $1.52_{\pm 0.42}$ | $1.14_{\pm 0.13}$ | $1.06_{\pm 0.08}$ | $-65.89_{\pm 29.35}$ | $0.95_{\pm 0.05}$ | $1.52_{\pm 0.42}$ | $1.14_{\pm 0.13}$ | $0.95_{\pm 0.05}$ |

Table 9: Overall results for GTOPX unconstrained tasks with 128 solutions and 100th percentile evaluations. In this case, 0% of the values are missing near the worst value and another 10% near the optimal value. Details are the same as Table 5.

| Steps | | | t = 50 | | | | | t = 100 | | | | | t = 150 | | |
|---|---|---|---|---|---|---|---|---|---|---|---|---|---|---|---|
| Task | | | | | | | | GTOPX 2 | | | | | | | |
| $f(x^*_{\mathrm{OFF}})$ | | | | | | | | $-136.28_{\pm 0.39}$ | | | | | | | |
| Metric | FS | SI | OI | SO | SO$_\omega$ | FS | SI | OI | SO | SO$_\omega$ | FS | SI | OI | SO | SO$_\omega$ |
| ARCOO | $-60.61_{\pm 4.74}$ | $0.94_{\pm 0.02}$ | $1.88_{\pm 0.04}$ | $1.25_{\pm 0.02}$ | $1.41_{\pm 0.03}$ | $-61.58_{\pm 5.63}$ | $0.94_{\pm 0.02}$ | $1.88_{\pm 0.04}$ | $1.26_{\pm 0.03}$ | $1.13_{\pm 0.03}$ | $-61.98_{\pm 7.44}$ | $0.94_{\pm 0.02}$ | $1.88_{\pm 0.05}$ | $1.25_{\pm 0.03}$ | $0.94_{\pm 0.02}$ |
| BO | $-90.59_{\pm 10.22}$ | $0.75_{\pm 0.02}$ | $1.45_{\pm 0.04}$ | $0.99_{\pm 0.02}$ | $1.11_{\pm 0.02}$ | $-104.88_{\pm 15.27}$ | $0.74_{\pm 0.02}$ | $1.46_{\pm 0.03}$ | $0.98_{\pm 0.02}$ | $0.89_{\pm 0.02}$ | $-99.83_{\pm 17.44}$ | $0.73_{\pm 0.01}$ | $1.46_{\pm 0.03}$ | $0.97_{\pm 0.02}$ | $0.73_{\pm 0.01}$ |
| CBAS | $-136.28_{\pm 0.39}$ | $1.00_{\pm 0.00}$ | $1.00_{\pm 0.00}$ | $1.00_{\pm 0.00}$ | $1.00_{\pm 0.00}$ | $-136.28_{\pm 0.39}$ | $1.00_{\pm 0.00}$ | $1.00_{\pm 0.00}$ | $1.00_{\pm 0.00}$ | $1.00_{\pm 0.00}$ | $-136.28_{\pm 0.39}$ | $1.00_{\pm 0.00}$ | $1.00_{\pm 0.00}$ | $1.00_{\pm 0.00}$ | $1.00_{\pm 0.00}$ |
| CCDDEA | $-261.39_{\pm 341.74}$ | $-0.05_{\pm 0.98}$ | $-0.10_{\pm 1.97}$ | $-0.07_{\pm 1.31}$ | $-0.08_{\pm 1.48}$ | $-548.44_{\pm 524.23}$ | $-0.50_{\pm 0.73}$ | $-1.01_{\pm 1.47}$ | $-0.67_{\pm 0.98}$ | $-0.61_{\pm 0.88}$ | $-553.40_{\pm 532.57}$ | $-1.06_{\pm 1.53}$ | $-2.12_{\pm 3.05}$ | $-1.42_{\pm 2.03}$ | $-1.07_{\pm 1.55}$ |
| CMAES | $-52.80_{\pm 4.17}$ | $0.95_{\pm 0.02}$ | $1.91_{\pm 0.04}$ | $1.27_{\pm 0.02}$ | $1.43_{\pm 0.03}$ | - | - | - | - | - | - | - | - | - | - |
| TTDDEA | $-273.91_{\pm 100.21}$ | $-0.30_{\pm 0.66}$ | $-0.61_{\pm 1.32}$ | $-0.41_{\pm 0.88}$ | $-0.46_{\pm 0.99}$ | $-345.81_{\pm 201.75}$ | $-0.66_{\pm 0.92}$ | $-1.33_{\pm 1.85}$ | $-0.89_{\pm 1.23}$ | $-0.80_{\pm 1.11}$ | $-380.69_{\pm 244.62}$ | $-0.92_{\pm 1.24}$ | $-1.84_{\pm 2.47}$ | $-1.23_{\pm 1.65}$ | $-0.92_{\pm 1.24}$ |
| Trimentoring | $-55.34_{\pm 11.69}$ | $0.94_{\pm 0.03}$ | $1.88_{\pm 0.04}$ | $1.25_{\pm 0.03}$ | $1.41_{\pm 0.03}$ | $-57.41_{\pm 11.75}$ | $0.94_{\pm 0.03}$ | $1.89_{\pm 0.07}$ | $1.26_{\pm 0.04}$ | $1.14_{\pm 0.04}$ | $-60.35_{\pm 11.48}$ | $0.94_{\pm 0.03}$ | $1.88_{\pm 0.06}$ | $1.26_{\pm 0.04}$ | $0.94_{\pm 0.03}$ |
| Task | | | | | | | | GTOPX 3 | | | | | | | |
| $f(x^*_{\mathrm{OFF}})$ | | | | | | | | $-102.69_{\pm 0.49}$ | | | | | | | |
| ARCOO | $-47.51_{\pm 6.28}$ | $0.93_{\pm 0.03}$ | $1.81_{\pm 0.05}$ | $1.23_{\pm 0.03}$ | $1.38_{\pm 0.03}$ | $-42.07_{\pm 4.48}$ | $0.93_{\pm 0.01}$ | $1.83_{\pm 0.08}$ | $1.23_{\pm 0.04}$ | $1.12_{\pm 0.04}$ | $-41.91_{\pm 7.22}$ | $0.93_{\pm 0.01}$ | $1.85_{\pm 0.07}$ | $1.24_{\pm 0.04}$ | $0.93_{\pm 0.03}$ |
| BO | $-67.72_{\pm 10.67}$ | $0.77_{\pm 0.02}$ | $1.50_{\pm 0.03}$ | $1.02_{\pm 0.02}$ | $1.14_{\pm 0.02}$ | $-63.61_{\pm 13.13}$ | $0.76_{\pm 0.01}$ | $1.51_{\pm 0.02}$ | $1.01_{\pm 0.01}$ | $0.91_{\pm 0.01}$ | $-62.48_{\pm 10.85}$ | $0.75_{\pm 0.01}$ | $1.50_{\pm 0.02}$ | $1.00_{\pm 0.02}$ | $0.75_{\pm 0.02}$ |
| CBAS | $-102.69_{\pm 0.49}$ | $1.00_{\pm 0.00}$ | $1.00_{\pm 0.00}$ | $1.00_{\pm 0.00}$ | $1.00_{\pm 0.00}$ | $-102.69_{\pm 0.49}$ | $1.00_{\pm 0.00}$ | $1.00_{\pm 0.00}$ | $1.00_{\pm 0.00}$ | $1.00_{\pm 0.00}$ | $-102.69_{\pm 0.49}$ | $1.00_{\pm 0.00}$ | $1.00_{\pm 0.00}$ | $1.00_{\pm 0.00}$ | $1.00_{\pm 0.00}$ |
| CCDDEA | $-8.16e4_{\pm 2.12e5}$ | $-76.58_{\pm 164.61}$ | $-153.15_{\pm 329.21}$ | $-102.10_{\pm 219.47}$ | $-115.06_{\pm 247.32}$ | $-144.66_{\pm 360.61}$ | $-50.24_{\pm 92.13}$ | $-100.48_{\pm 184.25}$ | $-66.99_{\pm 122.84}$ | $-60.45_{\pm 110.85}$ | $-170.07_{\pm 71.39}$ | $-33.41_{\pm 61.21}$ | $-66.82_{\pm 122.42}$ | $-44.54_{\pm 81.61}$ | $-33.52_{\pm 61.41}$ |
| CMAES | $-46.30_{\pm 13.76}$ | $0.91_{\pm 0.04}$ | $1.82_{\pm 0.08}$ | $1.21_{\pm 0.05}$ | $1.37_{\pm 0.06}$ | - | - | - | - | - | - | - | - | - | - |
| TTDDEA | $-2.16e5_{\pm 5.67e5}$ | $-1.10e3_{\pm 1.59e3}$ | $-2.20e3_{\pm 3.16e3}$ | $-1.47e3_{\pm 2.11e3}$ | $-1.65e3_{\pm 2.37e3}$ | $-2.16e6_{\pm 4.18e6}$ | $-1.43e4_{\pm 1.45e4}$ | $-2.85e4_{\pm 2.90e4}$ | $-1.90e4_{\pm 1.93e4}$ | $-1.72e4_{\pm 1.74e4}$ | $-4.76e6_{\pm 5.88e6}$ | $-2.81e4_{\pm 2.71e4}$ | $-5.62e4_{\pm 5.41e4}$ | $-3.75e4_{\pm 3.61e4}$ | $-2.82e4_{\pm 2.72e4}$ |
| Trimentoring | $-39.81_{\pm 6.08}$ | $0.95_{\pm 0.02}$ | $1.88_{\pm 0.04}$ | $1.26_{\pm 0.03}$ | $1.42_{\pm 0.03}$ | $-41.31_{\pm 4.29}$ | $0.96_{\pm 0.02}$ | $1.91_{\pm 0.06}$ | $1.27_{\pm 0.03}$ | $1.15_{\pm 0.02}$ | $-45.87_{\pm 6.25}$ | $0.95_{\pm 0.02}$ | $1.90_{\pm 0.03}$ | $1.27_{\pm 0.02}$ | $0.95_{\pm 0.02}$ |
| Task | | | | | | | | GTOPX 4 | | | | | | | |
| $f(x^*_{\mathrm{OFF}})$ | | | | | | | | $-153.62_{\pm 0.60}$ | | | | | | | |
| ARCOO | $-133.19_{\pm 23.01}$ | $0.68_{\pm 0.09}$ | $1.33_{\pm 0.16}$ | $0.90_{\pm 0.12}$ | $1.01_{\pm 0.13}$ | $-139.82_{\pm 23.32}$ | $0.67_{\pm 0.12}$ | $1.31_{\pm 0.25}$ | $0.88_{\pm 0.16}$ | $0.80_{\pm 0.14}$ | $-129.09_{\pm 25.59}$ | $0.66_{\pm 0.10}$ | $1.31_{\pm 0.21}$ | $0.88_{\pm 0.14}$ | $0.66_{\pm 0.10}$ |
| BO | $-106.32_{\pm 15.55}$ | $0.74_{\pm 0.03}$ | $1.46_{\pm 0.05}$ | $0.98_{\pm 0.04}$ | $1.11_{\pm 0.04}$ | $-88.73_{\pm 17.87}$ | $0.74_{\pm 0.03}$ | $1.45_{\pm 0.03}$ | $0.98_{\pm 0.03}$ | $0.88_{\pm 0.03}$ | $-101.73_{\pm 27.37}$ | $0.73_{\pm 0.01}$ | $1.45_{\pm 0.02}$ | $0.97_{\pm 0.01}$ | $0.73_{\pm 0.01}$ |
| CBAS | $-153.62_{\pm 0.60}$ | $1.00_{\pm 0.00}$ | $1.00_{\pm 0.00}$ | $1.00_{\pm 0.00}$ | $1.00_{\pm 0.00}$ | $-153.62_{\pm 0.60}$ | $1.00_{\pm 0.00}$ | $1.00_{\pm 0.00}$ | $1.00_{\pm 0.00}$ | $1.00_{\pm 0.00}$ | $-153.62_{\pm 0.60}$ | $1.00_{\pm 0.00}$ | $1.00_{\pm 0.00}$ | $1.00_{\pm 0.00}$ | $1.00_{\pm 0.00}$ |
| CCDDEA | $-1.54e3_{\pm 1.94e3}$ | $-161.09_{\pm 224.57}$ | $-322.19_{\pm 449.14}$ | $-214.79_{\pm 299.43}$ | $-242.04_{\pm 337.42}$ | $-2.94e7_{\pm 5.77e7}$ | $-1.76e5_{\pm 3.95e5}$ | $-3.52e5_{\pm 7.90e5}$ | $-2.35e5_{\pm 5.27e5}$ | $-2.12e5_{\pm 4.76e5}$ | $-1.65e7_{\pm 3.13e7}$ | $-1.66e5_{\pm 2.80e5}$ | $-3.32e5_{\pm 5.72e5}$ | $-2.21e5_{\pm 3.82e5}$ | $-1.66e5_{\pm 2.87e5}$ |
| CMAES | $-88.94_{\pm 17.93}$ | $0.83_{\pm 0.04}$ | $1.67_{\pm 0.07}$ | $1.11_{\pm 0.05}$ | $1.25_{\pm 0.05}$ | - | - | - | - | - | - | - | - | - | - |
| TTDDEA | $-7.24e3_{\pm 1.85e4}$ | $-25.91_{\pm 69.48}$ | $-51.82_{\pm 138.98}$ | $-34.55_{\pm 92.64}$ | $-38.93_{\pm 104.59}$ | $-5.42e4_{\pm 1.43e5}$ | $-143.58_{\pm 380.40}$ | $-287.17_{\pm 760.80}$ | $-191.44_{\pm 507.20}$ | $-172.76_{\pm 457.69}$ | $-3.76e7_{\pm 9.95e7}$ | $-9.35e3_{\pm 2.47e4}$ | $-1.87e4_{\pm 4.95e4}$ | $-1.25e4_{\pm 3.30e4}$ | $-9.38e3_{\pm 2.48e4}$ |
| Trimentoring | $-122.98_{\pm 17.45}$ | $0.72_{\pm 0.11}$ | $1.44_{\pm 0.21}$ | $0.96_{\pm 0.14}$ | $1.08_{\pm 0.16}$ | $-127.20_{\pm 21.16}$ | $0.69_{\pm 0.09}$ | $1.37_{\pm 0.17}$ | $0.92_{\pm 0.11}$ | $0.83_{\pm 0.10}$ | $-109.58_{\pm 24.30}$ | $0.70_{\pm 0.08}$ | $1.40_{\pm 0.17}$ | $0.94_{\pm 0.11}$ | $0.70_{\pm 0.08}$ |
| Task | | | | | | | | GTOPX 6 | | | | | | | |
| $f(x^*_{\mathrm{OFF}})$ | | | | | | | | $-83.37_{\pm 0.25}$ | | | | | | | |
| ARCOO | $-56.34_{\pm 13.34}$ | $0.86_{\pm 0.06}$ | $1.72_{\pm 0.11}$ | $1.15_{\pm 0.08}$ | $1.29_{\pm 0.08}$ | $-59.09_{\pm 13.58}$ | $0.81_{\pm 0.10}$ | $1.61_{\pm 0.20}$ | $1.08_{\pm 0.14}$ | $0.97_{\pm 0.12}$ | $-60.24_{\pm 12.43}$ | $0.78_{\pm 0.11}$ | $1.57_{\pm 0.23}$ | $1.05_{\pm 0.15}$ | $0.79_{\pm 0.11}$ |
| BO | $-66.20_{\pm 5.47}$ | $0.76_{\pm 0.02}$ | $1.45_{\pm 0.04}$ | $0.99_{\pm 0.02}$ | $1.11_{\pm 0.02}$ | $-63.39_{\pm 8.76}$ | $0.73_{\pm 0.03}$ | $1.44_{\pm 0.04}$ | $0.97_{\pm 0.03}$ | $0.88_{\pm 0.03}$ | $-63.05_{\pm 14.39}$ | $0.72_{\pm 0.02}$ | $1.44_{\pm 0.04}$ | $0.96_{\pm 0.03}$ | $0.72_{\pm 0.02}$ |
| CBAS | $-83.37_{\pm 0.25}$ | $1.00_{\pm 0.00}$ | $1.00_{\pm 0.00}$ | $1.00_{\pm 0.00}$ | $1.00_{\pm 0.00}$ | $-83.37_{\pm 0.25}$ | $1.00_{\pm 0.00}$ | $1.00_{\pm 0.00}$ | $1.00_{\pm 0.00}$ | $1.00_{\pm 0.00}$ | $-83.37_{\pm 0.25}$ | $1.00_{\pm 0.00}$ | $1.00_{\pm 0.00}$ | $1.00_{\pm 0.00}$ | $1.00_{\pm 0.00}$ |
| CCDDEA | $-208.15_{\pm 166.83}$ | $-0.24_{\pm 0.92}$ | $-0.46_{\pm 1.03}$ | $-0.31_{\pm 0.69}$ | $-0.35_{\pm 0.78}$ | $-150.61_{\pm 65.70}$ | $-0.30_{\pm 0.40}$ | $-0.61_{\pm 0.79}$ | $-0.41_{\pm 0.53}$ | $-0.37_{\pm 0.48}$ | $-155.23_{\pm 60.73}$ | $-0.32_{\pm 0.35}$ | $-0.63_{\pm 0.71}$ | $-0.42_{\pm 0.47}$ | $-0.32_{\pm 0.35}$ |
| CMAES | $-39.46_{\pm 5.93}$ | $0.89_{\pm 0.03}$ | $1.78_{\pm 0.09}$ | $1.19_{\pm 0.04}$ | $1.34_{\pm 0.04}$ | - | - | - | - | - | - | - | - | - | - |
| TTDDEA | $-115.04_{\pm 32.62}$ | $0.38_{\pm 0.28}$ | $0.76_{\pm 0.56}$ | $0.51_{\pm 0.37}$ | $0.57_{\pm 0.42}$ | $-124.12_{\pm 28.70}$ | $0.21_{\pm 0.34}$ | $0.42_{\pm 0.68}$ | $0.28_{\pm 0.46}$ | $0.26_{\pm 0.41}$ | $-298.76_{\pm 373.94}$ | $0.05_{\pm 0.39}$ | $0.09_{\pm 0.79}$ | $0.06_{\pm 0.53}$ | $0.05_{\pm 0.40}$ |
| Trimentoring | $-48.53_{\pm 13.68}$ | $0.90_{\pm 0.06}$ | $1.67_{\pm 0.27}$ | $1.15_{\pm 0.08}$ | $1.28_{\pm 0.12}$ | $-49.40_{\pm 13.95}$ | $0.89_{\pm 0.06}$ | $1.66_{\pm 0.27}$ | $1.15_{\pm 0.08}$ | $1.05_{\pm 0.06}$ | $-51.21_{\pm 14.03}$ | $0.89_{\pm 0.07}$ | $1.65_{\pm 0.26}$ | $1.14_{\pm 0.08}$ | $0.89_{\pm 0.06}$ |

Table 10: Overall results for GTOPX unconstrained tasks with 128 solutions and 100th percentile evaluations. In this case, 10% of the values are missing near the worst value and another 40% near the optimal value. Details are the same as Table 5.

**GTOPX 2** — $f(x^*_{OFF}) = -275.27_{\pm 2.34}$

| Metric | FS (t=50) | SI | OI | SO | SO$_\omega$ | FS (t=100) | SI | OI | SO | SO$_\omega$ | FS (t=150) | SI | OI | SO | SO$_\omega$ |
|---|---|---|---|---|---|---|---|---|---|---|---|---|---|---|---|
| ARCOO | -78.85$_{\pm13.45}$ | 0.94$_{\pm0.01}$ | 1.85$_{\pm0.05}$ | 1.25$_{\pm0.02}$ | 1.40$_{\pm0.03}$ | -72.33$_{\pm10.06}$ | 0.95$_{\pm0.02}$ | 1.89$_{\pm0.04}$ | 1.27$_{\pm0.02}$ | 1.14$_{\pm0.02}$ | -68.61$_{\pm9.09}$ | 0.96$_{\pm0.02}$ | 1.91$_{\pm0.03}$ | 1.28$_{\pm0.02}$ | 0.96$_{\pm0.02}$ |
| BO | -95.93$_{\pm21.71}$ | 0.90$_{\pm0.01}$ | 1.78$_{\pm0.01}$ | 1.19$_{\pm0.01}$ | 1.34$_{\pm0.01}$ | -99.74$_{\pm25.59}$ | 0.90$_{\pm0.01}$ | 1.79$_{\pm0.01}$ | 1.20$_{\pm0.01}$ | 1.08$_{\pm0.01}$ | -105.80$_{\pm17.91}$ | 0.90$_{\pm0.01}$ | 1.79$_{\pm0.01}$ | 1.19$_{\pm0.01}$ | 0.90$_{\pm0.01}$ |
| CBAS | -275.27$_{\pm2.34}$ | 1.00$_{\pm0.00}$ | 1.00$_{\pm0.00}$ | 1.00$_{\pm0.00}$ | 1.00$_{\pm0.00}$ | -275.27$_{\pm2.34}$ | 1.00$_{\pm0.00}$ | 1.00$_{\pm0.00}$ | 1.00$_{\pm0.00}$ | 1.00$_{\pm0.00}$ | -275.27$_{\pm2.34}$ | 1.00$_{\pm0.00}$ | 1.00$_{\pm0.00}$ | 1.00$_{\pm0.00}$ | 1.00$_{\pm0.00}$ |
| CCDDEA | -250.42$_{\pm197.30}$ | 0.63$_{\pm0.11}$ | 1.25$_{\pm0.22}$ | 0.84$_{\pm0.14}$ | 0.94$_{\pm0.16}$ | -876.09$_{\pm601.36}$ | 0.24$_{\pm0.42}$ | 0.49$_{\pm0.83}$ | 0.33$_{\pm0.55}$ | 0.29$_{\pm0.50}$ | -714.28$_{\pm322.50}$ | -0.05$_{\pm0.71}$ | -0.10$_{\pm1.43}$ | -0.07$_{\pm0.95}$ | -0.05$_{\pm0.72}$ |
| CMAES | -59.06$_{\pm3.33}$ | 0.98$_{\pm0.01}$ | 1.96$_{\pm0.02}$ | 1.30$_{\pm0.01}$ | 1.47$_{\pm0.01}$ | - | - | - | - | - | - | - | - | - | - |
| TTDDEA | -230.79$_{\pm56.94}$ | 0.69$_{\pm0.11}$ | 1.38$_{\pm0.21}$ | 0.92$_{\pm0.14}$ | 1.03$_{\pm0.16}$ | -206.66$_{\pm39.54}$ | 0.66$_{\pm0.09}$ | 1.32$_{\pm0.18}$ | 0.88$_{\pm0.12}$ | 0.79$_{\pm0.11}$ | -218.44$_{\pm58.28}$ | 0.66$_{\pm0.09}$ | 1.32$_{\pm0.17}$ | 0.88$_{\pm0.11}$ | 0.66$_{\pm0.09}$ |
| Trimentoring | -111.87$_{\pm88.20}$ | 0.98$_{\pm0.01}$ | 1.83$_{\pm0.31}$ | 1.27$_{\pm0.10}$ | 1.41$_{\pm0.15}$ | -111.05$_{\pm88.36}$ | 0.99$_{\pm0.01}$ | 1.84$_{\pm0.12}$ | 1.27$_{\pm0.10}$ | 1.16$_{\pm0.06}$ | -109.44$_{\pm89.20}$ | 0.99$_{\pm0.01}$ | 1.85$_{\pm0.32}$ | 1.27$_{\pm0.10}$ | 0.99$_{\pm0.01}$ |

**GTOPX 3** — $f(x^*_{OFF}) = -228.84_{\pm3.46}$

| Metric | FS (t=50) | SI | OI | SO | SO$_\omega$ | FS (t=100) | SI | OI | SO | SO$_\omega$ | FS (t=150) | SI | OI | SO | SO$_\omega$ |
|---|---|---|---|---|---|---|---|---|---|---|---|---|---|---|---|
| ARCOO | -51.89$_{\pm9.13}$ | 0.96$_{\pm0.01}$ | 1.92$_{\pm0.02}$ | 1.28$_{\pm0.01}$ | 1.44$_{\pm0.02}$ | -49.87$_{\pm7.99}$ | 0.97$_{\pm0.01}$ | 1.93$_{\pm0.02}$ | 1.29$_{\pm0.01}$ | 1.16$_{\pm0.01}$ | -46.84$_{\pm11.74}$ | 0.97$_{\pm0.01}$ | 1.94$_{\pm0.02}$ | 1.29$_{\pm0.01}$ | 0.97$_{\pm0.01}$ |
| BO | -66.55$_{\pm5.86}$ | 0.92$_{\pm0.01}$ | 1.81$_{\pm0.02}$ | 1.22$_{\pm0.01}$ | 1.37$_{\pm0.01}$ | -64.97$_{\pm16.39}$ | 0.92$_{\pm0.01}$ | 1.82$_{\pm0.01}$ | 1.22$_{\pm0.01}$ | 1.10$_{\pm0.01}$ | -69.40$_{\pm8.65}$ | 0.91$_{\pm0.01}$ | 1.82$_{\pm0.01}$ | 1.21$_{\pm0.01}$ | 0.91$_{\pm0.01}$ |
| CBAS | -228.84$_{\pm3.46}$ | 1.00$_{\pm0.00}$ | 1.00$_{\pm0.00}$ | 1.00$_{\pm0.00}$ | 1.00$_{\pm0.00}$ | -228.84$_{\pm3.46}$ | 1.00$_{\pm0.00}$ | 1.00$_{\pm0.00}$ | 1.00$_{\pm0.00}$ | 1.00$_{\pm0.00}$ | -228.84$_{\pm3.46}$ | 1.00$_{\pm0.00}$ | 1.00$_{\pm0.00}$ | 1.00$_{\pm0.00}$ | 1.00$_{\pm0.00}$ |
| CCDDEA | -107.15$_{\pm22.76}$ | -1.10$_{\pm3.13}$ | -2.21$_{\pm6.24}$ | -1.47$_{\pm4.17}$ | -1.66$_{\pm4.69}$ | -352.86$_{\pm479.62}$ | -0.35$_{\pm1.59}$ | -0.70$_{\pm3.19}$ | -0.46$_{\pm2.13}$ | -0.42$_{\pm1.92}$ | -352.63$_{\pm479.41}$ | -0.19$_{\pm1.24}$ | -0.37$_{\pm2.49}$ | -0.25$_{\pm1.66}$ | -0.19$_{\pm1.25}$ |
| CMAES | -41.24$_{\pm4.62}$ | 0.97$_{\pm0.01}$ | 1.95$_{\pm0.02}$ | 1.30$_{\pm0.01}$ | 1.46$_{\pm0.02}$ | - | - | - | - | - | - | - | - | - | - |
| TTDDEA | -748.95$_{\pm981.96}$ | 0.04$_{\pm1.69}$ | 0.09$_{\pm3.38}$ | 0.06$_{\pm2.25}$ | 0.07$_{\pm2.54}$ | -8.31e3$_{\pm9.17e3}$ | -17.52$_{\pm40.61}$ | -35.03$_{\pm81.21}$ | -23.35$_{\pm54.14}$ | -21.08$_{\pm48.86}$ | -2.54e6$_{\pm6.40e6}$ | -301.89$_{\pm609.79}$ | -603.79$_{\pm1.22e3}$ | -402.52$_{\pm813.05}$ | -302.90$_{\pm611.81}$ |
| Trimentoring | -66.31$_{\pm63.01}$ | 0.90$_{\pm0.18}$ | 1.75$_{\pm0.50}$ | 1.19$_{\pm0.29}$ | 1.33$_{\pm0.34}$ | -65.50$_{\pm63.26}$ | 0.91$_{\pm0.19}$ | 1.76$_{\pm0.51}$ | 1.19$_{\pm0.29}$ | 1.08$_{\pm0.25}$ | -66.86$_{\pm62.43}$ | 0.91$_{\pm0.19}$ | 1.76$_{\pm0.51}$ | 1.19$_{\pm0.29}$ | 0.91$_{\pm0.19}$ |

**GTOPX 4** — $f(x^*_{OFF}) = -322.50_{\pm1.08}$

| Metric | FS (t=50) | SI | OI | SO | SO$_\omega$ | FS (t=100) | SI | OI | SO | SO$_\omega$ | FS (t=150) | SI | OI | SO | SO$_\omega$ |
|---|---|---|---|---|---|---|---|---|---|---|---|---|---|---|---|
| ARCOO | -125.40$_{\pm25.46}$ | 0.93$_{\pm0.03}$ | 1.86$_{\pm0.05}$ | 1.24$_{\pm0.04}$ | 1.40$_{\pm0.04}$ | -136.19$_{\pm30.99}$ | 0.91$_{\pm0.04}$ | 1.82$_{\pm0.07}$ | 1.21$_{\pm0.05}$ | 1.09$_{\pm0.04}$ | -135.38$_{\pm31.37}$ | 0.90$_{\pm0.04}$ | 1.80$_{\pm0.08}$ | 1.20$_{\pm0.05}$ | 0.90$_{\pm0.04}$ |
| BO | -96.36$_{\pm19.43}$ | 0.91$_{\pm0.01}$ | 1.80$_{\pm0.01}$ | 1.21$_{\pm0.01}$ | 1.36$_{\pm0.01}$ | -119.84$_{\pm19.55}$ | 0.90$_{\pm0.01}$ | 1.80$_{\pm0.01}$ | 1.20$_{\pm0.01}$ | 1.09$_{\pm0.01}$ | -94.93$_{\pm18.82}$ | 0.90$_{\pm0.01}$ | 1.80$_{\pm0.01}$ | 1.20$_{\pm0.01}$ | 0.90$_{\pm0.01}$ |
| CBAS | -322.50$_{\pm1.08}$ | 1.00$_{\pm0.00}$ | 1.00$_{\pm0.00}$ | 1.00$_{\pm0.00}$ | 1.00$_{\pm0.00}$ | -322.50$_{\pm1.08}$ | 1.00$_{\pm0.00}$ | 1.00$_{\pm0.00}$ | 1.00$_{\pm0.00}$ | 1.00$_{\pm0.00}$ | -322.50$_{\pm1.08}$ | 1.00$_{\pm0.00}$ | 1.00$_{\pm0.00}$ | 1.00$_{\pm0.00}$ | 1.00$_{\pm0.00}$ |
| CCDDEA | -433.31$_{\pm296.79}$ | -5.61$_{\pm4.97}$ | -11.21$_{\pm17.95}$ | -7.47$_{\pm11.96}$ | -8.42$_{\pm13.48}$ | -2.55e5$_{\pm3.70e5}$ | -3.26e3$_{\pm8.21e3}$ | -6.52e3$_{\pm1.64e4}$ | -4.35e3$_{\pm1.09e4}$ | -3.92e3$_{\pm9.87e3}$ | -9.28e5$_{\pm1.75e6}$ | -2.70e3$_{\pm5.52e3}$ | -5.41e3$_{\pm1.10e4}$ | -3.61e3$_{\pm7.36e3}$ | -2.71e3$_{\pm5.56e3}$ |
| CMAES | -90.12$_{\pm7.02}$ | 0.95$_{\pm0.02}$ | 1.89$_{\pm0.04}$ | 1.26$_{\pm0.03}$ | 1.42$_{\pm0.03}$ | - | - | - | - | - | - | - | - | - | - |
| TTDDEA | -2.11e4$_{\pm3.66e4}$ | -24.80$_{\pm41.83}$ | -49.60$_{\pm83.66}$ | -33.07$_{\pm55.77}$ | -37.26$_{\pm62.83}$ | -6.36e4$_{\pm1.13e5}$ | -58.34$_{\pm106.27}$ | -116.68$_{\pm212.54}$ | -77.79$_{\pm141.69}$ | -70.19$_{\pm127.86}$ | -4.27e4$_{\pm7.54e4}$ | -67.84$_{\pm119.62}$ | -135.68$_{\pm239.24}$ | -90.46$_{\pm159.49}$ | -68.07$_{\pm120.02}$ |
| Trimentoring | -123.24$_{\pm31.25}$ | 0.91$_{\pm0.08}$ | 1.81$_{\pm0.10}$ | 1.21$_{\pm0.06}$ | 1.36$_{\pm0.07}$ | -127.16$_{\pm27.12}$ | 0.90$_{\pm0.05}$ | 1.80$_{\pm0.10}$ | 1.20$_{\pm0.07}$ | 1.08$_{\pm0.06}$ | -139.15$_{\pm37.70}$ | 0.89$_{\pm0.08}$ | 1.78$_{\pm0.10}$ | 1.19$_{\pm0.06}$ | 0.90$_{\pm0.05}$ |

**GTOPX 6** — $f(x^*_{OFF}) = -142.09_{\pm0.39}$

| Metric | FS (t=50) | SI | OI | SO | SO$_\omega$ | FS (t=100) | SI | OI | SO | SO$_\omega$ | FS (t=150) | SI | OI | SO | SO$_\omega$ |
|---|---|---|---|---|---|---|---|---|---|---|---|---|---|---|---|
| ARCOO | -51.10$_{\pm9.35}$ | 0.92$_{\pm0.04}$ | 1.82$_{\pm0.08}$ | 1.22$_{\pm0.05}$ | 1.38$_{\pm0.05}$ | -52.52$_{\pm16.62}$ | 0.93$_{\pm0.02}$ | 1.86$_{\pm0.05}$ | 1.24$_{\pm0.03}$ | 1.12$_{\pm0.03}$ | -54.14$_{\pm12.30}$ | 0.93$_{\pm0.04}$ | 1.86$_{\pm0.07}$ | 1.24$_{\pm0.05}$ | 0.93$_{\pm0.04}$ |
| BO | -56.65$_{\pm7.99}$ | 0.88$_{\pm0.01}$ | 1.72$_{\pm0.04}$ | 1.17$_{\pm0.01}$ | 1.31$_{\pm0.02}$ | -57.12$_{\pm9.11}$ | 0.88$_{\pm0.02}$ | 1.73$_{\pm0.04}$ | 1.17$_{\pm0.02}$ | 1.05$_{\pm0.02}$ | -57.19$_{\pm16.23}$ | 0.87$_{\pm0.02}$ | 1.73$_{\pm0.03}$ | 1.16$_{\pm0.02}$ | 0.87$_{\pm0.02}$ |
| CBAS | -142.09$_{\pm0.39}$ | 1.00$_{\pm0.00}$ | 1.00$_{\pm0.00}$ | 1.00$_{\pm0.00}$ | 1.00$_{\pm0.00}$ | -142.09$_{\pm0.39}$ | 1.00$_{\pm0.00}$ | 1.00$_{\pm0.00}$ | 1.00$_{\pm0.00}$ | 1.00$_{\pm0.00}$ | -142.09$_{\pm0.39}$ | 1.00$_{\pm0.00}$ | 1.00$_{\pm0.00}$ | 1.00$_{\pm0.00}$ | 1.00$_{\pm0.00}$ |
| CCDDEA | -210.91$_{\pm117.18}$ | 0.27$_{\pm0.24}$ | 0.53$_{\pm0.48}$ | 0.35$_{\pm0.32}$ | 0.40$_{\pm0.36}$ | -725.78$_{\pm686.74}$ | -0.66$_{\pm1.21}$ | -1.32$_{\pm2.43}$ | -0.88$_{\pm1.62}$ | -0.79$_{\pm1.46}$ | -683.07$_{\pm757.16}$ | -1.55$_{\pm2.49}$ | -3.10$_{\pm4.98}$ | -2.07$_{\pm3.32}$ | -1.56$_{\pm2.50}$ |
| CMAES | -43.42$_{\pm8.00}$ | 0.96$_{\pm0.01}$ | 1.91$_{\pm0.02}$ | 1.27$_{\pm0.01}$ | 1.44$_{\pm0.02}$ | - | - | - | - | - | - | - | - | - | - |
| TTDDEA | -108.37$_{\pm27.46}$ | 0.74$_{\pm0.05}$ | 1.49$_{\pm0.10}$ | 0.99$_{\pm0.06}$ | 1.12$_{\pm0.07}$ | -105.79$_{\pm28.12}$ | 0.71$_{\pm0.09}$ | 1.42$_{\pm0.18}$ | 0.95$_{\pm0.12}$ | 0.85$_{\pm0.11}$ | -111.04$_{\pm29.45}$ | 0.70$_{\pm0.11}$ | 1.39$_{\pm0.21}$ | 0.93$_{\pm0.14}$ | 0.70$_{\pm0.11}$ |
| Trimentoring | -51.10$_{\pm9.15}$ | 0.92$_{\pm0.06}$ | 1.83$_{\pm0.11}$ | 1.22$_{\pm0.07}$ | 1.38$_{\pm0.08}$ | -52.09$_{\pm12.41}$ | 0.92$_{\pm0.06}$ | 1.83$_{\pm0.11}$ | 1.22$_{\pm0.07}$ | 1.10$_{\pm0.07}$ | -52.11$_{\pm12.45}$ | 0.92$_{\pm0.06}$ | 1.83$_{\pm0.11}$ | 1.22$_{\pm0.07}$ | 0.92$_{\pm0.06}$ |

Table 11: Overall results for GTOPX unconstrained tasks with 128 solutions and 100th percentile evaluations. In this case, 20% of the values are missing near the worst value and another 30% near the optimal value. Details are the same as Table 5.

**GTOPX 2** — $f(x^*_{OFF}) = -218.12_{\pm1.78}$

| Metric | FS (t=50) | SI | OI | SO | SO$_\omega$ | FS (t=100) | SI | OI | SO | SO$_\omega$ | FS (t=150) | SI | OI | SO | SO$_\omega$ |
|---|---|---|---|---|---|---|---|---|---|---|---|---|---|---|---|
| ARCOO | -76.31$_{\pm7.92}$ | 0.91$_{\pm0.02}$ | 1.81$_{\pm0.04}$ | 1.21$_{\pm0.03}$ | 1.36$_{\pm0.03}$ | -76.53$_{\pm10.59}$ | 0.94$_{\pm0.02}$ | 1.87$_{\pm0.03}$ | 1.25$_{\pm0.02}$ | 1.12$_{\pm0.02}$ | -76.81$_{\pm14.25}$ | 0.94$_{\pm0.02}$ | 1.88$_{\pm0.03}$ | 1.25$_{\pm0.02}$ | 0.94$_{\pm0.02}$ |
| BO | -102.46$_{\pm11.83}$ | 0.87$_{\pm0.02}$ | 1.72$_{\pm0.02}$ | 1.16$_{\pm0.02}$ | 1.30$_{\pm0.02}$ | -93.14$_{\pm16.12}$ | 0.86$_{\pm0.01}$ | 1.72$_{\pm0.01}$ | 1.14$_{\pm0.02}$ | 1.03$_{\pm0.02}$ | -116.17$_{\pm29.39}$ | 0.86$_{\pm0.02}$ | 1.72$_{\pm0.05}$ | 1.15$_{\pm0.02}$ | 0.86$_{\pm0.02}$ |
| CBAS | -218.12$_{\pm1.78}$ | 1.00$_{\pm0.00}$ | 1.00$_{\pm0.00}$ | 1.00$_{\pm0.00}$ | 1.00$_{\pm0.00}$ | -218.12$_{\pm1.78}$ | 1.00$_{\pm0.00}$ | 1.00$_{\pm0.00}$ | 1.00$_{\pm0.00}$ | 1.00$_{\pm0.00}$ | -218.12$_{\pm1.78}$ | 1.00$_{\pm0.00}$ | 1.00$_{\pm0.00}$ | 1.00$_{\pm0.00}$ | 1.00$_{\pm0.00}$ |
| CCDDEA | -183.76$_{\pm39.39}$ | 0.52$_{\pm0.21}$ | 1.04$_{\pm0.42}$ | 0.70$_{\pm0.28}$ | 0.78$_{\pm0.32}$ | -1.21e3$_{\pm1.25e3}$ | -2.69$_{\pm5.87}$ | -5.38$_{\pm11.73}$ | -3.59$_{\pm7.82}$ | -3.24$_{\pm7.06}$ | -800.03$_{\pm418.25}$ | -2.39$_{\pm4.25}$ | -4.77$_{\pm8.49}$ | -3.18$_{\pm5.66}$ | -2.40$_{\pm4.26}$ |
| CMAES | -62.10$_{\pm3.43}$ | 0.97$_{\pm0.01}$ | 1.94$_{\pm0.01}$ | 1.29$_{\pm0.01}$ | 1.46$_{\pm0.01}$ | - | - | - | - | - | - | - | - | - | - |
| TTDDEA | -262.69$_{\pm79.36}$ | 0.54$_{\pm0.18}$ | 1.07$_{\pm0.37}$ | 0.71$_{\pm0.24}$ | 0.80$_{\pm0.28}$ | -263.17$_{\pm82.54}$ | 0.44$_{\pm0.21}$ | 0.89$_{\pm0.42}$ | 0.59$_{\pm0.28}$ | 0.53$_{\pm0.25}$ | -257.60$_{\pm84.27}$ | 0.42$_{\pm0.24}$ | 0.83$_{\pm0.48}$ | 0.55$_{\pm0.32}$ | 0.42$_{\pm0.24}$ |
| Trimentoring | -100.10$_{\pm69.53}$ | 0.98$_{\pm0.01}$ | 1.71$_{\pm0.41}$ | 1.23$_{\pm0.13}$ | 1.35$_{\pm0.20}$ | -100.42$_{\pm69.27}$ | 0.98$_{\pm0.02}$ | 1.72$_{\pm0.41}$ | 1.23$_{\pm0.13}$ | 1.13$_{\pm0.08}$ | -99.55$_{\pm69.76}$ | 0.98$_{\pm0.02}$ | 1.72$_{\pm0.41}$ | 1.23$_{\pm0.13}$ | 0.99$_{\pm0.01}$ |

**GTOPX 3** — $f(x^*_{OFF}) = -171.54_{\pm0.98}$

| Metric | FS (t=50) | SI | OI | SO | SO$_\omega$ | FS (t=100) | SI | OI | SO | SO$_\omega$ | FS (t=150) | SI | OI | SO | SO$_\omega$ |
|---|---|---|---|---|---|---|---|---|---|---|---|---|---|---|---|
| ARCOO | -50.09$_{\pm4.81}$ | 0.95$_{\pm0.02}$ | 1.87$_{\pm0.02}$ | 1.26$_{\pm0.02}$ | 1.42$_{\pm0.02}$ | -52.61$_{\pm12.37}$ | 0.94$_{\pm0.02}$ | 1.88$_{\pm0.02}$ | 1.26$_{\pm0.02}$ | 1.13$_{\pm0.02}$ | -53.14$_{\pm13.24}$ | 0.94$_{\pm0.02}$ | 1.89$_{\pm0.04}$ | 1.26$_{\pm0.03}$ | 0.95$_{\pm0.02}$ |
| BO | -74.46$_{\pm19.16}$ | 0.87$_{\pm0.01}$ | 1.72$_{\pm0.03}$ | 1.16$_{\pm0.02}$ | 1.30$_{\pm0.02}$ | -60.06$_{\pm12.66}$ | 0.87$_{\pm0.02}$ | 1.73$_{\pm0.02}$ | 1.16$_{\pm0.02}$ | 1.05$_{\pm0.02}$ | -69.97$_{\pm20.80}$ | 0.87$_{\pm0.01}$ | 1.73$_{\pm0.02}$ | 1.15$_{\pm0.01}$ | 0.87$_{\pm0.01}$ |
| CBAS | -171.54$_{\pm0.81}$ | 1.00$_{\pm0.00}$ | 1.00$_{\pm0.00}$ | 1.00$_{\pm0.00}$ | 1.00$_{\pm0.00}$ | -171.54$_{\pm0.81}$ | 1.00$_{\pm0.00}$ | 1.00$_{\pm0.00}$ | 1.00$_{\pm0.00}$ | 1.00$_{\pm0.00}$ | -171.54$_{\pm0.81}$ | 1.00$_{\pm0.00}$ | 1.00$_{\pm0.00}$ | 1.00$_{\pm0.00}$ | 1.00$_{\pm0.00}$ |
| CCDDEA | -229.31$_{\pm317.76}$ | 0.11$_{\pm0.92}$ | 0.21$_{\pm1.83}$ | 0.14$_{\pm1.22}$ | 0.16$_{\pm1.38}$ | -504.52$_{\pm541.28}$ | -0.48$_{\pm1.81}$ | -0.95$_{\pm3.61}$ | -0.63$_{\pm2.41}$ | -0.57$_{\pm2.17}$ | -609.80$_{\pm808.94}$ | -0.67$_{\pm2.17}$ | -1.35$_{\pm4.24}$ | -0.90$_{\pm2.83}$ | -0.68$_{\pm2.15}$ |
| CMAES | -41.89$_{\pm4.74}$ | 0.97$_{\pm0.02}$ | 1.93$_{\pm0.04}$ | 1.29$_{\pm0.03}$ | 1.45$_{\pm0.03}$ | - | - | - | - | - | - | - | - | - | - |
| TTDDEA | -2.44e3$_{\pm5.78e3}$ | -8.43$_{\pm22.93}$ | -16.87$_{\pm45.85}$ | -11.25$_{\pm30.57}$ | -12.67$_{\pm34.45}$ | -2.69e4$_{\pm5.37e4}$ | -29.12$_{\pm59.55}$ | -58.24$_{\pm119.09}$ | -38.83$_{\pm79.40}$ | -35.04$_{\pm71.65}$ | -7.67e5$_{\pm1.58e6}$ | -722.24$_{\pm1.78e3}$ | -1.44e3$_{\pm3.56e3}$ | -962.99$_{\pm2375.48}$ | -724.64$_{\pm1787.53}$ |
| Trimentoring | -42.04$_{\pm5.43}$ | 0.98$_{\pm0.01}$ | 1.96$_{\pm0.03}$ | 1.31$_{\pm0.02}$ | 1.48$_{\pm0.02}$ | -42.68$_{\pm5.54}$ | 0.98$_{\pm0.02}$ | 1.97$_{\pm0.03}$ | 1.31$_{\pm0.02}$ | 1.18$_{\pm0.02}$ | -42.80$_{\pm5.61}$ | 0.98$_{\pm0.02}$ | 1.97$_{\pm0.03}$ | 1.31$_{\pm0.02}$ | 0.99$_{\pm0.02}$ |

**GTOPX 4** — $f(x^*_{OFF}) = -241.61_{\pm1.39}$

| Metric | FS (t=50) | SI | OI | SO | SO$_\omega$ | FS (t=100) | SI | OI | SO | SO$_\omega$ | FS (t=150) | SI | OI | SO | SO$_\omega$ |
|---|---|---|---|---|---|---|---|---|---|---|---|---|---|---|---|
| ARCOO | -93.66$_{\pm15.68}$ | 0.92$_{\pm0.03}$ | 1.83$_{\pm0.05}$ | 1.23$_{\pm0.04}$ | 1.38$_{\pm0.04}$ | -113.65$_{\pm13.83}$ | 0.91$_{\pm0.03}$ | 1.81$_{\pm0.05}$ | 1.21$_{\pm0.04}$ | 1.09$_{\pm0.03}$ | -131.27$_{\pm26.97}$ | 0.89$_{\pm0.03}$ | 1.78$_{\pm0.07}$ | 1.19$_{\pm0.05}$ | 0.89$_{\pm0.03}$ |
| BO | -106.41$_{\pm25.10}$ | 0.87$_{\pm0.02}$ | 1.69$_{\pm0.00}$ | 1.14$_{\pm0.02}$ | 1.28$_{\pm0.02}$ | -109.70$_{\pm21.89}$ | 0.87$_{\pm0.01}$ | 1.70$_{\pm0.03}$ | 1.15$_{\pm0.01}$ | 1.04$_{\pm0.01}$ | -107.88$_{\pm19.62}$ | 0.85$_{\pm0.01}$ | 1.71$_{\pm0.02}$ | 1.14$_{\pm0.02}$ | 0.86$_{\pm0.01}$ |
| CBAS | -241.61$_{\pm1.39}$ | 1.00$_{\pm0.00}$ | 1.00$_{\pm0.00}$ | 1.00$_{\pm0.00}$ | 1.00$_{\pm0.00}$ | -241.61$_{\pm1.39}$ | 1.00$_{\pm0.00}$ | 1.00$_{\pm0.00}$ | 1.00$_{\pm0.00}$ | 1.00$_{\pm0.00}$ | -241.61$_{\pm1.39}$ | 1.00$_{\pm0.00}$ | 1.00$_{\pm0.00}$ | 1.00$_{\pm0.00}$ | 1.00$_{\pm0.00}$ |
| CCDDEA | -756.53$_{\pm602.24}$ | -9.29$_{\pm14.96}$ | -18.58$_{\pm23.96}$ | -12.39$_{\pm15.97}$ | -13.96$_{\pm24.00}$ | -1.86e5$_{\pm2.66e5}$ | -163.06$_{\pm213.53}$ | -326.12$_{\pm426.65}$ | -217.41$_{\pm284.45}$ | -196.19$_{\pm256.67}$ | -1.86e5$_{\pm2.66e5}$ | -344.86$_{\pm471.77}$ | -689.73$_{\pm943.55}$ | -459.82$_{\pm629.02}$ | -346.01$_{\pm473.33}$ |
| CMAES | -99.45$_{\pm18.58}$ | 0.90$_{\pm0.02}$ | 1.80$_{\pm0.04}$ | 1.20$_{\pm0.03}$ | 1.35$_{\pm0.03}$ | - | - | - | - | - | - | - | - | - | - |
| TTDDEA | -996.45$_{\pm1.95e3}$ | 0.36$_{\pm0.71}$ | 0.71$_{\pm1.42}$ | 0.47$_{\pm0.95}$ | 0.53$_{\pm1.07}$ | -1.75e3$_{\pm3.94e3}$ | -1.84$_{\pm4.26}$ | -3.68$_{\pm12.52}$ | -2.46$_{\pm8.35}$ | -2.22$_{\pm7.53}$ | -3288.01$_{\pm7952.34}$ | -3.11$_{\pm9.63}$ | -6.23$_{\pm19.03}$ | -4.15$_{\pm12.69}$ | -3.12$_{\pm9.55}$ |
| Trimentoring | -147.56$_{\pm41.87}$ | 0.88$_{\pm0.06}$ | 1.64$_{\pm0.24}$ | 1.14$_{\pm0.07}$ | 1.26$_{\pm0.11}$ | -145.76$_{\pm42.70}$ | 0.86$_{\pm0.07}$ | 1.60$_{\pm0.24}$ | 1.11$_{\pm0.08}$ | 1.01$_{\pm0.06}$ | -146.68$_{\pm42.23}$ | 0.85$_{\pm0.08}$ | 1.58$_{\pm0.24}$ | 1.10$_{\pm0.08}$ | 0.85$_{\pm0.08}$ |

**GTOPX 6** — $f(x^*_{OFF}) = -121.48_{\pm0.38}$

| Metric | FS (t=50) | SI | OI | SO | SO$_\omega$ | FS (t=100) | SI | OI | SO | SO$_\omega$ | FS (t=150) | SI | OI | SO | SO$_\omega$ |
|---|---|---|---|---|---|---|---|---|---|---|---|---|---|---|---|
| ARCOO | -55.29$_{\pm10.53}$ | 0.90$_{\pm0.04}$ | 1.77$_{\pm0.08}$ | 1.19$_{\pm0.06}$ | 1.34$_{\pm0.06}$ | -61.99$_{\pm25.18}$ | 0.89$_{\pm0.04}$ | 1.77$_{\pm0.12}$ | 1.18$_{\pm0.08}$ | 1.07$_{\pm0.07}$ | -62.17$_{\pm24.79}$ | 0.87$_{\pm0.09}$ | 1.75$_{\pm0.17}$ | 1.16$_{\pm0.12}$ | 0.88$_{\pm0.09}$ |
| BO | -64.20$_{\pm10.41}$ | 0.83$_{\pm0.01}$ | 1.66$_{\pm0.02}$ | 1.12$_{\pm0.01}$ | 1.26$_{\pm0.01}$ | -63.39$_{\pm19.30}$ | 0.83$_{\pm0.04}$ | 1.67$_{\pm0.02}$ | 1.11$_{\pm0.01}$ | 1.00$_{\pm0.01}$ | -55.89$_{\pm13.86}$ | 0.83$_{\pm0.01}$ | 1.67$_{\pm0.02}$ | 1.11$_{\pm0.01}$ | 0.83$_{\pm0.01}$ |
| CBAS | -121.48$_{\pm0.38}$ | 1.00$_{\pm0.00}$ | 1.00$_{\pm0.00}$ | 1.00$_{\pm0.00}$ | 1.00$_{\pm0.00}$ | -121.48$_{\pm0.38}$ | 1.00$_{\pm0.00}$ | 1.00$_{\pm0.00}$ | 1.00$_{\pm0.00}$ | 1.00$_{\pm0.00}$ | -121.48$_{\pm0.38}$ | 1.00$_{\pm0.00}$ | 1.00$_{\pm0.00}$ | 1.00$_{\pm0.00}$ | 1.00$_{\pm0.00}$ |
| CCDDEA | -126.69$_{\pm31.53}$ | 0.28$_{\pm0.16}$ | 0.54$_{\pm0.33}$ | 0.37$_{\pm0.22}$ | 0.41$_{\pm0.25}$ | -564.31$_{\pm481.67}$ | -0.39$_{\pm0.77}$ | -0.77$_{\pm1.54}$ | -0.52$_{\pm1.03}$ | -0.46$_{\pm0.93}$ | -562.87$_{\pm483.47}$ | -1.12$_{\pm1.61}$ | -2.24$_{\pm3.22}$ | -1.49$_{\pm2.15}$ | -1.12$_{\pm1.62}$ |
| CMAES | -42.83$_{\pm4.96}$ | 0.95$_{\pm0.05}$ | 1.89$_{\pm0.11}$ | 1.26$_{\pm0.07}$ | 1.42$_{\pm0.08}$ | - | - | - | - | - | - | - | - | - | - |
| TTDDEA | -85.26$_{\pm15.26}$ | 0.75$_{\pm0.06}$ | 1.51$_{\pm0.12}$ | 1.01$_{\pm0.08}$ | 1.13$_{\pm0.09}$ | -90.75$_{\pm51.82}$ | 0.73$_{\pm0.10}$ | 1.45$_{\pm0.19}$ | 0.97$_{\pm0.13}$ | 0.87$_{\pm0.11}$ | -97.17$_{\pm54.11}$ | 0.70$_{\pm0.13}$ | 1.41$_{\pm0.26}$ | 0.94$_{\pm0.17}$ | 0.71$_{\pm0.13}$ |
| Trimentoring | -77.85$_{\pm55.10}$ | 0.95$_{\pm0.05}$ | 1.53$_{\pm0.43}$ | 1.14$_{\pm0.14}$ | 1.24$_{\pm0.21}$ | -78.78$_{\pm54.36}$ | 0.94$_{\pm0.06}$ | 1.51$_{\pm0.42}$ | 1.12$_{\pm0.13}$ | 1.05$_{\pm0.08}$ | -79.98$_{\pm53.40}$ | 0.94$_{\pm0.06}$ | 1.50$_{\pm0.41}$ | 1.12$_{\pm0.13}$ | 0.94$_{\pm0.06}$ |

Table 12: Overall results for GTOPX constrained tasks with 128 solutions and 100th percentile evaluations. In this case, 25% of the values are missing near the worst value and another 25% near the optimal value. Details are the same as Table 5.

| Steps | t = 50 | | | | | t = 100 | | | | | t = 150 | | | | |
|---|---|---|---|---|---|---|---|---|---|---|---|---|---|---|---|
| Task $f(x^*_{OFF})$ | GTOPX 2 $-195.44_{\pm1.38}$ | | | | | | | | | | | | | | |
| ARCOO | $-68.68_{\pm6.13}$ | $0.92_{\pm0.01}$ | $1.76_{\pm0.08}$ | $1.21_{\pm0.02}$ | $1.35_{\pm0.02}$ | $-61.39_{\pm12.98}$ | $0.93_{\pm0.01}$ | $1.84_{\pm0.04}$ | $1.23_{\pm0.02}$ | $1.11_{\pm0.02}$ | $-58.89_{\pm9.36}$ | $0.93_{\pm0.02}$ | $1.86_{\pm0.04}$ | $1.24_{\pm0.02}$ | $0.94_{\pm0.02}$ |
| BO | $-100.67_{\pm17.26}$ | $0.86_{\pm0.02}$ | $1.66_{\pm0.02}$ | $1.13_{\pm0.02}$ | $1.27_{\pm0.02}$ | $-89.30_{\pm18.91}$ | $0.85_{\pm0.01}$ | $1.67_{\pm0.02}$ | $1.13_{\pm0.01}$ | $1.02_{\pm0.01}$ | $-96.04_{\pm11.15}$ | $0.84_{\pm0.01}$ | $1.68_{\pm0.02}$ | $1.12_{\pm0.01}$ | $0.84_{\pm0.01}$ |
| CBAS | $-195.44_{\pm1.38}$ | $1.00_{\pm0.00}$ | $1.00_{\pm0.00}$ | $1.00_{\pm0.00}$ | $1.00_{\pm0.00}$ | $-195.44_{\pm1.38}$ | $1.00_{\pm0.00}$ | $1.00_{\pm0.00}$ | $1.00_{\pm0.00}$ | $1.00_{\pm0.00}$ | $-195.44_{\pm1.38}$ | $1.00_{\pm0.00}$ | $1.00_{\pm0.00}$ | $1.00_{\pm0.00}$ | $1.00_{\pm0.00}$ |
| CCDDEA | $-166.15_{\pm41.96}$ | $0.62_{\pm0.15}$ | $1.23_{\pm0.30}$ | $0.83_{\pm0.26}$ | $0.93_{\pm0.23}$ | $-1.03e3_{\pm871.51}$ | $-0.11_{\pm0.58}$ | $-0.21_{\pm1.16}$ | $-0.14_{\pm0.77}$ | $-0.13_{\pm0.70}$ | $-1.03e3_{\pm871.50}$ | $-1.01_{\pm1.55}$ | $-2.01_{\pm3.10}$ | $-1.34_{\pm2.06}$ | $-1.01_{\pm1.55}$ |
| CMAES | $-57.06_{\pm6.42}$ | $0.96_{\pm0.02}$ | $1.93_{\pm0.04}$ | $1.29_{\pm0.03}$ | $1.45_{\pm0.03}$ | - | - | - | - | - | - | - | - | - | - |
| TTDDEA | $-245.65_{\pm47.93}$ | $0.45_{\pm0.11}$ | $0.91_{\pm0.22}$ | $0.60_{\pm0.15}$ | $0.68_{\pm0.17}$ | $-241.67_{\pm35.09}$ | $0.37_{\pm0.13}$ | $0.74_{\pm0.26}$ | $0.49_{\pm0.17}$ | $0.44_{\pm0.16}$ | $-242.57_{\pm40.75}$ | $0.35_{\pm0.12}$ | $0.71_{\pm0.24}$ | $0.47_{\pm0.16}$ | $0.35_{\pm0.12}$ |
| Trimentoring | $-75.48_{\pm45.69}$ | $0.98_{\pm0.01}$ | $1.83_{\pm0.31}$ | $1.26_{\pm0.10}$ | $1.40_{\pm0.15}$ | $-74.16_{\pm46.26}$ | $0.98_{\pm0.01}$ | $1.83_{\pm0.32}$ | $1.26_{\pm0.10}$ | $1.15_{\pm0.06}$ | $-73.79_{\pm46.43}$ | $0.98_{\pm0.01}$ | $1.84_{\pm0.32}$ | $1.27_{\pm0.10}$ | $0.98_{\pm0.01}$ |
| Task $f(x^*_{OFF})$ | GTOPX 3 $-151.85_{\pm0.66}$ | | | | | | | | | | | | | | |
| ARCOO | $-47.38_{\pm7.64}$ | $0.94_{\pm0.02}$ | $1.84_{\pm0.08}$ | $1.25_{\pm0.02}$ | $1.40_{\pm0.04}$ | $-51.31_{\pm8.89}$ | $0.94_{\pm0.01}$ | $1.86_{\pm0.03}$ | $1.25_{\pm0.02}$ | $1.12_{\pm0.02}$ | $-49.03_{\pm11.43}$ | $0.93_{\pm0.02}$ | $1.86_{\pm0.03}$ | $1.24_{\pm0.02}$ | $0.93_{\pm0.02}$ |
| BO | $-57.83_{\pm12.15}$ | $0.86_{\pm0.01}$ | $1.70_{\pm0.03}$ | $1.14_{\pm0.02}$ | $1.28_{\pm0.02}$ | $-62.75_{\pm16.92}$ | $0.85_{\pm0.01}$ | $1.70_{\pm0.01}$ | $1.14_{\pm0.01}$ | $1.03_{\pm0.01}$ | $-62.88_{\pm16.44}$ | $0.85_{\pm0.01}$ | $1.70_{\pm0.01}$ | $1.13_{\pm0.01}$ | $0.85_{\pm0.01}$ |
| CBAS | $-151.85_{\pm0.66}$ | $1.00_{\pm0.00}$ | $1.00_{\pm0.00}$ | $1.00_{\pm0.00}$ | $1.00_{\pm0.00}$ | $-151.85_{\pm0.66}$ | $1.00_{\pm0.00}$ | $1.00_{\pm0.00}$ | $1.00_{\pm0.00}$ | $1.00_{\pm0.00}$ | $-151.85_{\pm0.66}$ | $1.00_{\pm0.00}$ | $1.00_{\pm0.00}$ | $1.00_{\pm0.00}$ | $1.00_{\pm0.00}$ |
| CCDDEA | $-5.92e4_{\pm1.56e3}$ | $-5.04_{\pm3.09}$ | $-10.09_{\pm16.17}$ | $-6.72_{\pm10.78}$ | $-7.57_{\pm12.15}$ | $-212.15_{\pm85.56}$ | $-4.35_{\pm9.29}$ | $-8.70_{\pm18.57}$ | $-5.80_{\pm12.38}$ | $-5.24_{\pm11.17}$ | $-214.82_{\pm93.52}$ | $-2.82_{\pm6.26}$ | $-5.64_{\pm12.52}$ | $-3.76_{\pm8.34}$ | $-2.83_{\pm6.28}$ |
| CMAES | $-39.41_{\pm4.45}$ | $0.97_{\pm0.01}$ | $1.94_{\pm0.02}$ | $1.29_{\pm0.01}$ | $1.45_{\pm0.02}$ | - | - | - | - | - | - | - | - | - | - |
| TTDDEA | $-157.10_{\pm92.73}$ | $0.67_{\pm0.21}$ | $1.34_{\pm0.42}$ | $0.90_{\pm0.28}$ | $1.01_{\pm0.32}$ | $-837.59_{\pm836.31}$ | $0.25_{\pm0.53}$ | $0.50_{\pm1.06}$ | $0.33_{\pm0.71}$ | $0.30_{\pm0.64}$ | $-1.47e6_{\pm3.67e6}$ | $-133.47_{\pm295.80}$ | $-266.94_{\pm391.60}$ | $-177.96_{\pm394.40}$ | $-133.91_{\pm296.78}$ |
| Trimentoring | $-109.04_{\pm30.67}$ | $0.99_{\pm0.02}$ | $1.48_{\pm0.47}$ | $1.15_{\pm0.15}$ | $1.23_{\pm0.23}$ | $-108.97_{\pm50.74}$ | $0.99_{\pm0.02}$ | $1.48_{\pm0.47}$ | $1.15_{\pm0.15}$ | $1.09_{\pm0.09}$ | $-107.85_{\pm32.05}$ | $0.99_{\pm0.02}$ | $1.48_{\pm0.48}$ | $1.15_{\pm0.15}$ | $0.99_{\pm0.02}$ |
| Task $f(x^*_{OFF})$ | GTOPX 4 $-215.74_{\pm1.18}$ | | | | | | | | | | | | | | |
| ARCOO | $-87.64_{\pm14.50}$ | $0.93_{\pm0.02}$ | $1.85_{\pm0.06}$ | $1.24_{\pm0.04}$ | $1.40_{\pm0.04}$ | $-103.86_{\pm15.77}$ | $0.92_{\pm0.02}$ | $1.85_{\pm0.05}$ | $1.23_{\pm0.03}$ | $1.11_{\pm0.03}$ | $-111.28_{\pm18.64}$ | $0.91_{\pm0.03}$ | $1.83_{\pm0.05}$ | $1.22_{\pm0.03}$ | $0.92_{\pm0.03}$ |
| BO | $-115.19_{\pm15.73}$ | $0.85_{\pm0.02}$ | $1.66_{\pm0.04}$ | $1.13_{\pm0.02}$ | $1.26_{\pm0.03}$ | $-116.43_{\pm14.72}$ | $0.84_{\pm0.02}$ | $1.67_{\pm0.03}$ | $1.12_{\pm0.02}$ | $1.01_{\pm0.02}$ | $-115.82_{\pm18.31}$ | $0.83_{\pm0.01}$ | $1.67_{\pm0.03}$ | $1.11_{\pm0.02}$ | $0.84_{\pm0.01}$ |
| CBAS | $-215.74_{\pm1.18}$ | $1.00_{\pm0.00}$ | $1.00_{\pm0.00}$ | $1.00_{\pm0.00}$ | $1.00_{\pm0.00}$ | $-215.74_{\pm1.18}$ | $1.00_{\pm0.00}$ | $1.00_{\pm0.00}$ | $1.00_{\pm0.00}$ | $1.00_{\pm0.00}$ | $-215.74_{\pm1.18}$ | $1.00_{\pm0.00}$ | $1.00_{\pm0.00}$ | $1.00_{\pm0.00}$ | $1.00_{\pm0.00}$ |
| CCDDEA | $-379.37_{\pm357.65}$ | $-5.48_{\pm11.42}$ | $-10.96_{\pm22.81}$ | $-7.31_{\pm15.22}$ | $-8.23_{\pm17.15}$ | $-1.83e5_{\pm2.88e5}$ | $-279.28_{\pm376.27}$ | $-558.50_{\pm752.54}$ | $-372.37_{\pm501.70}$ | $-336.02_{\pm482.73}$ | $-9.80e4_{\pm1.61e5}$ | $-382.91_{\pm517.72}$ | $-765.82_{\pm1.04e3}$ | $-510.55_{\pm690.29}$ | $-384.18_{\pm519.44}$ |
| CMAES | $-95.48_{\pm19.86}$ | $0.91_{\pm0.03}$ | $1.83_{\pm0.06}$ | $1.22_{\pm0.04}$ | $1.37_{\pm0.04}$ | - | - | - | - | - | - | - | - | - | - |
| TTDDEA | $-259.66_{\pm132.15}$ | $0.43_{\pm0.44}$ | $0.87_{\pm0.88}$ | $0.58_{\pm0.59}$ | $0.65_{\pm0.66}$ | $-903.36_{\pm1.81e3}$ | $-0.10_{\pm1.64}$ | $-0.19_{\pm3.27}$ | $-0.13_{\pm2.18}$ | $-0.12_{\pm1.97}$ | $-308.87_{\pm133.56}$ | $-0.40_{\pm2.39}$ | $-0.80_{\pm4.77}$ | $-0.53_{\pm3.18}$ | $-0.40_{\pm2.39}$ |
| Trimentoring | $-139.48_{\pm57.84}$ | $0.94_{\pm0.05}$ | $1.63_{\pm0.38}$ | $1.17_{\pm0.12}$ | $1.29_{\pm0.18}$ | $-142.48_{\pm56.15}$ | $0.94_{\pm0.06}$ | $1.63_{\pm0.38}$ | $1.17_{\pm0.12}$ | $1.08_{\pm0.08}$ | $-137.73_{\pm59.21}$ | $0.94_{\pm0.06}$ | $1.64_{\pm0.39}$ | $1.18_{\pm0.13}$ | $0.94_{\pm0.06}$ |
| Task $f(x^*_{OFF})$ | GTOPX 6 $-112.11_{\pm0.33}$ | | | | | | | | | | | | | | |
| ARCOO | $-49.43_{\pm10.25}$ | $0.91_{\pm0.03}$ | $1.79_{\pm0.07}$ | $1.21_{\pm0.04}$ | $1.36_{\pm0.04}$ | $-54.77_{\pm23.78}$ | $0.89_{\pm0.06}$ | $1.78_{\pm0.12}$ | $1.19_{\pm0.08}$ | $1.07_{\pm0.07}$ | $-56.30_{\pm11.92}$ | $0.87_{\pm0.07}$ | $1.75_{\pm0.15}$ | $1.16_{\pm0.10}$ | $0.88_{\pm0.07}$ |
| BO | $-61.97_{\pm9.22}$ | $0.84_{\pm0.01}$ | $1.63_{\pm0.03}$ | $1.11_{\pm0.02}$ | $1.24_{\pm0.02}$ | $-60.91_{\pm10.49}$ | $0.82_{\pm0.02}$ | $1.63_{\pm0.03}$ | $1.09_{\pm0.02}$ | $0.99_{\pm0.02}$ | $-60.49_{\pm10.67}$ | $0.82_{\pm0.01}$ | $1.64_{\pm0.03}$ | $1.09_{\pm0.02}$ | $0.82_{\pm0.01}$ |
| CBAS | $-112.11_{\pm0.33}$ | $1.00_{\pm0.00}$ | $1.00_{\pm0.00}$ | $1.00_{\pm0.00}$ | $1.00_{\pm0.00}$ | $-112.11_{\pm0.33}$ | $1.00_{\pm0.00}$ | $1.00_{\pm0.00}$ | $1.00_{\pm0.00}$ | $1.00_{\pm0.00}$ | $-112.11_{\pm0.33}$ | $1.00_{\pm0.00}$ | $1.00_{\pm0.00}$ | $1.00_{\pm0.00}$ | $1.00_{\pm0.00}$ |
| CCDDEA | $-150.90_{\pm49.16}$ | $0.27_{\pm0.21}$ | $0.53_{\pm0.42}$ | $0.35_{\pm0.28}$ | $0.40_{\pm0.32}$ | $-3.53e3_{\pm3.35e3}$ | $-6.12_{\pm14.96}$ | $-12.23_{\pm20.72}$ | $-8.15_{\pm13.82}$ | $-7.36_{\pm12.47}$ | $-3.52e3_{\pm3.36e3}$ | $-13.26_{\pm21.85}$ | $-26.52_{\pm43.69}$ | $-17.68_{\pm29.13}$ | $-13.30_{\pm21.92}$ |
| CMAES | $-46.42_{\pm3.38}$ | $0.92_{\pm0.03}$ | $1.84_{\pm0.06}$ | $1.23_{\pm0.04}$ | $1.38_{\pm0.04}$ | - | - | - | - | - | - | - | - | - | - |
| TTDDEA | $-132.26_{\pm51.65}$ | $0.44_{\pm0.29}$ | $0.88_{\pm0.58}$ | $0.59_{\pm0.39}$ | $0.66_{\pm0.43}$ | $-129.93_{\pm40.44}$ | $0.38_{\pm0.33}$ | $0.75_{\pm0.67}$ | $0.50_{\pm0.45}$ | $0.45_{\pm0.40}$ | $-154.89_{\pm70.24}$ | $0.34_{\pm0.34}$ | $0.67_{\pm0.68}$ | $0.45_{\pm0.45}$ | $0.34_{\pm0.34}$ |
| Trimentoring | $-82.74_{\pm30.03}$ | $0.85_{\pm0.18}$ | $1.45_{\pm0.41}$ | $1.05_{\pm0.22}$ | $1.15_{\pm0.26}$ | $-83.07_{\pm29.69}$ | $0.86_{\pm0.18}$ | $1.46_{\pm0.42}$ | $1.06_{\pm0.22}$ | $0.98_{\pm0.19}$ | $-81.71_{\pm29.12}$ | $0.86_{\pm0.18}$ | $1.47_{\pm0.42}$ | $1.06_{\pm0.21}$ | $0.86_{\pm0.18}$ |

Table 13: Overall results for GTOPX unconstrained tasks with 128 solutions and 100th percentile evaluations. In this case, 30% of the values are missing near the worst value and another 20% near the optimal value. Details are the same as Table 5.

| Steps | t = 50 | | | | | t = 100 | | | | | t = 150 | | | | |
|---|---|---|---|---|---|---|---|---|---|---|---|---|---|---|---|
| Task $f(x^*_{OFF})$ | GTOPX 2 $-175.10_{\pm0.75}$ | | | | | | | | | | | | | | |
| Metric | FS | SI | OI | SO | SO$_\smile$ | FS | SI | OI | SO | SO$_\smile$ | FS | SI | OI | SO | SO$_\smile$ |
| ARCOO | $-79.61_{\pm6.27}$ | $0.91_{\pm0.03}$ | $1.75_{\pm0.08}$ | $1.20_{\pm0.04}$ | $1.34_{\pm0.04}$ | $-75.65_{\pm10.82}$ | $0.92_{\pm0.02}$ | $1.83_{\pm0.04}$ | $1.22_{\pm0.03}$ | $1.10_{\pm0.03}$ | $-73.50_{\pm13.24}$ | $0.92_{\pm0.03}$ | $1.84_{\pm0.05}$ | $1.23_{\pm0.04}$ | $0.92_{\pm0.03}$ |
| BO | $-104.20_{\pm17.73}$ | $0.83_{\pm0.02}$ | $1.64_{\pm0.02}$ | $1.10_{\pm0.02}$ | $1.24_{\pm0.02}$ | $-95.34_{\pm14.71}$ | $0.82_{\pm0.01}$ | $1.63_{\pm0.02}$ | $1.09_{\pm0.01}$ | $0.99_{\pm0.01}$ | $-89.75_{\pm21.91}$ | $0.82_{\pm0.01}$ | $1.64_{\pm0.02}$ | $1.09_{\pm0.02}$ | $0.82_{\pm0.01}$ |
| CBAS | $-175.10_{\pm0.75}$ | $1.00_{\pm0.00}$ | $1.00_{\pm0.00}$ | $1.00_{\pm0.00}$ | $1.00_{\pm0.00}$ | $-175.10_{\pm0.75}$ | $1.00_{\pm0.00}$ | $1.00_{\pm0.00}$ | $1.00_{\pm0.00}$ | $1.00_{\pm0.00}$ | $-175.10_{\pm0.75}$ | $1.00_{\pm0.00}$ | $1.00_{\pm0.00}$ | $1.00_{\pm0.00}$ | $1.00_{\pm0.00}$ |
| CCDDEA | $-155.42_{\pm36.31}$ | $0.35_{\pm0.08}$ | $0.69_{\pm0.17}$ | $0.46_{\pm0.11}$ | $0.52_{\pm0.13}$ | $-1303.29_{\pm1614.43}$ | $-0.91_{\pm2.25}$ | $-1.82_{\pm4.50}$ | $-1.21_{\pm3.00}$ | $-1.09_{\pm2.71}$ | $-1533.95_{\pm2140.00}$ | $-2.32_{\pm4.51}$ | $-4.63_{\pm9.02}$ | $-3.09_{\pm6.02}$ | $-2.32_{\pm4.53}$ |
| CMAES | $-59.22_{\pm9.81}$ | $0.96_{\pm0.02}$ | $1.91_{\pm0.04}$ | $1.27_{\pm0.03}$ | $1.44_{\pm0.03}$ | - | - | - | - | - | - | - | - | - | - |
| TTDDEA | $-266.71_{\pm45.97}$ | $0.29_{\pm0.21}$ | $0.57_{\pm0.42}$ | $0.38_{\pm0.28}$ | $0.43_{\pm0.32}$ | $-259.04_{\pm30.65}$ | $0.19_{\pm0.17}$ | $0.39_{\pm0.34}$ | $0.26_{\pm0.23}$ | $0.23_{\pm0.20}$ | $-259.49_{\pm27.72}$ | $0.17_{\pm0.15}$ | $0.33_{\pm0.30}$ | $0.22_{\pm0.20}$ | $0.17_{\pm0.15}$ |
| Trimentoring | $-91.41_{\pm48.58}$ | $0.92_{\pm0.14}$ | $1.69_{\pm0.38}$ | $1.18_{\pm0.70}$ | $1.31_{\pm0.23}$ | $-90.28_{\pm49.46}$ | $0.91_{\pm0.13}$ | $1.69_{\pm0.36}$ | $1.17_{\pm0.20}$ | $1.07_{\pm0.17}$ | $-88.73_{\pm50.34}$ | $0.91_{\pm0.15}$ | $1.69_{\pm0.39}$ | $1.17_{\pm0.20}$ | $0.91_{\pm0.15}$ |
| Task $f(x^*_{OFF})$ | GTOPX 3 $-134.83_{\pm0.96}$ | | | | | | | | | | | | | | |
| ARCOO | $-50.66_{\pm5.39}$ | $0.92_{\pm0.04}$ | $1.81_{\pm0.06}$ | $1.22_{\pm0.04}$ | $1.37_{\pm0.04}$ | $-48.03_{\pm10.76}$ | $0.92_{\pm0.03}$ | $1.84_{\pm0.06}$ | $1.23_{\pm0.04}$ | $1.11_{\pm0.04}$ | $-48.00_{\pm6.80}$ | $0.92_{\pm0.03}$ | $1.84_{\pm0.07}$ | $1.23_{\pm0.04}$ | $0.93_{\pm0.03}$ |
| BO | $-62.14_{\pm9.46}$ | $0.85_{\pm0.01}$ | $1.66_{\pm0.02}$ | $1.13_{\pm0.02}$ | $1.26_{\pm0.02}$ | $-78.64_{\pm22.22}$ | $0.83_{\pm0.01}$ | $1.66_{\pm0.02}$ | $1.11_{\pm0.02}$ | $1.00_{\pm0.02}$ | $-61.53_{\pm7.28}$ | $0.83_{\pm0.01}$ | $1.67_{\pm0.02}$ | $1.11_{\pm0.01}$ | $0.83_{\pm0.01}$ |
| CBAS | $-134.83_{\pm0.96}$ | $1.00_{\pm0.00}$ | $1.00_{\pm0.00}$ | $1.00_{\pm0.00}$ | $1.00_{\pm0.00}$ | $-134.83_{\pm0.96}$ | $1.00_{\pm0.00}$ | $1.00_{\pm0.00}$ | $1.00_{\pm0.00}$ | $1.00_{\pm0.00}$ | $-134.83_{\pm0.96}$ | $1.00_{\pm0.00}$ | $1.00_{\pm0.00}$ | $1.00_{\pm0.00}$ | $1.00_{\pm0.00}$ |
| CCDDEA | $-121.47_{\pm24.84}$ | $-4.97_{\pm20.24}$ | $-9.94_{\pm20.26}$ | $-6.63_{\pm13.90}$ | $-7.47_{\pm15.21}$ | $-323.57_{\pm296.66}$ | $-2.52_{\pm5.38}$ | $-5.05_{\pm10.75}$ | $-3.37_{\pm7.17}$ | $-3.04_{\pm6.47}$ | $-323.55_{\pm296.77}$ | $-1.92_{\pm4.21}$ | $-3.84_{\pm8.42}$ | $-2.56_{\pm5.61}$ | $-1.93_{\pm4.22}$ |
| CMAES | $-41.09_{\pm4.73}$ | $0.96_{\pm0.02}$ | $1.92_{\pm0.03}$ | $1.28_{\pm0.02}$ | $1.44_{\pm0.02}$ | - | - | - | - | - | - | - | - | - | - |
| TTDDEA | $-164.40_{\pm129.44}$ | $0.42_{\pm0.59}$ | $0.84_{\pm1.19}$ | $0.56_{\pm0.79}$ | $0.63_{\pm0.89}$ | $-2.25e3_{\pm3.21e3}$ | $-0.96_{\pm2.72}$ | $-1.92_{\pm5.45}$ | $-1.28_{\pm3.63}$ | $-1.16_{\pm3.28}$ | $-2.26e6_{\pm5.85e6}$ | $-1.33e3_{\pm3.40e3}$ | $-2.66e3_{\pm6.80e3}$ | $-1.77e3_{\pm4.55e3}$ | $-1.33e3_{\pm3.41e3}$ |
| Trimentoring | $-45.31_{\pm8.61}$ | $0.96_{\pm0.02}$ | $1.92_{\pm0.08}$ | $1.28_{\pm0.03}$ | $1.44_{\pm0.04}$ | $-45.15_{\pm9.38}$ | $0.96_{\pm0.03}$ | $1.92_{\pm0.06}$ | $1.28_{\pm0.04}$ | $1.15_{\pm0.04}$ | $-44.40_{\pm7.99}$ | $0.96_{\pm0.03}$ | $1.92_{\pm0.06}$ | $1.28_{\pm0.04}$ | $0.96_{\pm0.03}$ |
| Task $f(x^*_{OFF})$ | GTOPX 4 $-194.03_{\pm0.88}$ | | | | | | | | | | | | | | |
| ARCOO | $-93.43_{\pm14.16}$ | $0.91_{\pm0.03}$ | $1.81_{\pm0.06}$ | $1.21_{\pm0.04}$ | $1.37_{\pm0.04}$ | $-103.01_{\pm13.99}$ | $0.91_{\pm0.03}$ | $1.81_{\pm0.05}$ | $1.21_{\pm0.03}$ | $1.09_{\pm0.03}$ | $-108.81_{\pm16.69}$ | $0.90_{\pm0.04}$ | $1.80_{\pm0.07}$ | $1.20_{\pm0.05}$ | $0.90_{\pm0.04}$ |
| BO | $-106.97_{\pm22.75}$ | $0.81_{\pm0.02}$ | $1.59_{\pm0.02}$ | $1.07_{\pm0.02}$ | $1.20_{\pm0.02}$ | $-115.69_{\pm39.77}$ | $0.80_{\pm0.01}$ | $1.59_{\pm0.02}$ | $1.07_{\pm0.02}$ | $0.96_{\pm0.01}$ | $-98.74_{\pm20.33}$ | $0.80_{\pm0.01}$ | $1.60_{\pm0.03}$ | $1.06_{\pm0.02}$ | $0.80_{\pm0.01}$ |
| CBAS | $-194.03_{\pm0.88}$ | $1.00_{\pm0.00}$ | $1.00_{\pm0.00}$ | $1.00_{\pm0.00}$ | $1.00_{\pm0.00}$ | $-194.03_{\pm0.88}$ | $1.00_{\pm0.00}$ | $1.00_{\pm0.00}$ | $1.00_{\pm0.00}$ | $1.00_{\pm0.00}$ | $-194.03_{\pm0.88}$ | $1.00_{\pm0.00}$ | $1.00_{\pm0.00}$ | $1.00_{\pm0.00}$ | $1.00_{\pm0.00}$ |
| CCDDEA | $-290.06_{\pm173.60}$ | $-29.74_{\pm65.19}$ | $-58.51_{\pm130.44}$ | $-39.41_{\pm86.93}$ | $-44.29_{\pm97.97}$ | $-2.04e5_{\pm4.07e5}$ | $-349.98_{\pm695.07}$ | $-699.96_{\pm1390.15}$ | $-466.64_{\pm926.76}$ | $-421.09_{\pm836.30}$ | $-3.22e6_{\pm7.84e6}$ | $-2.93e3_{\pm5.96e3}$ | $-5.86e3_{\pm1.19e4}$ | $-3.91e3_{\pm7.96e3}$ | $-2.94e3_{\pm5.98e3}$ |
| CMAES | $-95.89_{\pm19.66}$ | $0.88_{\pm0.05}$ | $1.76_{\pm0.10}$ | $1.17_{\pm0.07}$ | $1.32_{\pm0.07}$ | - | - | - | - | - | - | - | - | - | - |
| TTDDEA | $-228.13_{\pm40.47}$ | $0.52_{\pm0.19}$ | $1.03_{\pm0.37}$ | $0.69_{\pm0.25}$ | $0.77_{\pm0.28}$ | $-265.16_{\pm51.47}$ | $0.41_{\pm0.18}$ | $0.81_{\pm0.36}$ | $0.54_{\pm0.24}$ | $0.49_{\pm0.22}$ | $-256.06_{\pm67.40}$ | $0.36_{\pm0.19}$ | $0.72_{\pm0.37}$ | $0.48_{\pm0.25}$ | $0.36_{\pm0.19}$ |
| Trimentoring | $-139.75_{\pm53.78}$ | $0.82_{\pm0.12}$ | $1.40_{\pm0.26}$ | $1.01_{\pm0.09}$ | $1.11_{\pm0.11}$ | $-143.47_{\pm58.60}$ | $0.83_{\pm0.12}$ | $1.41_{\pm0.26}$ | $1.02_{\pm0.09}$ | $0.94_{\pm0.09}$ | $-146.76_{\pm30.50}$ | $0.83_{\pm0.13}$ | $1.41_{\pm0.27}$ | $1.02_{\pm0.09}$ | $0.83_{\pm0.12}$ |
| Task $f(x^*_{OFF})$ | GTOPX 6 $-102.96_{\pm0.29}$ | | | | | | | | | | | | | | |
| ARCOO | $-49.14_{\pm16.86}$ | $0.87_{\pm0.04}$ | $1.73_{\pm0.08}$ | $1.16_{\pm0.05}$ | $1.30_{\pm0.05}$ | $-59.69_{\pm18.16}$ | $0.86_{\pm0.07}$ | $1.72_{\pm0.14}$ | $1.15_{\pm0.09}$ | $1.04_{\pm0.08}$ | $-64.64_{\pm25.22}$ | $0.85_{\pm0.09}$ | $1.70_{\pm0.18}$ | $1.13_{\pm0.12}$ | $0.85_{\pm0.09}$ |
| BO | $-60.24_{\pm15.25}$ | $0.81_{\pm0.01}$ | $1.61_{\pm0.02}$ | $1.08_{\pm0.02}$ | $1.22_{\pm0.02}$ | $-61.30_{\pm11.46}$ | $0.80_{\pm0.01}$ | $1.61_{\pm0.03}$ | $1.07_{\pm0.02}$ | $0.97_{\pm0.02}$ | $-58.64_{\pm10.77}$ | $0.80_{\pm0.01}$ | $1.60_{\pm0.03}$ | $1.07_{\pm0.02}$ | $0.81_{\pm0.01}$ |
| CBAS | $-102.96_{\pm0.29}$ | $1.00_{\pm0.00}$ | $1.00_{\pm0.00}$ | $1.00_{\pm0.00}$ | $1.00_{\pm0.00}$ | $-102.96_{\pm0.29}$ | $1.00_{\pm0.00}$ | $1.00_{\pm0.00}$ | $1.00_{\pm0.00}$ | $1.00_{\pm0.00}$ | $-102.96_{\pm0.29}$ | $1.00_{\pm0.00}$ | $1.00_{\pm0.00}$ | $1.00_{\pm0.00}$ | $1.00_{\pm0.00}$ |
| CCDDEA | $-124.64_{\pm37.88}$ | $0.37_{\pm0.12}$ | $0.73_{\pm0.24}$ | $0.49_{\pm0.16}$ | $0.55_{\pm0.18}$ | $-683.70_{\pm786.10}$ | $-0.75_{\pm1.68}$ | $-1.49_{\pm3.36}$ | $-0.99_{\pm2.24}$ | $-0.90_{\pm2.02}$ | $-1.13e3_{\pm1.20e3}$ | $-2.86_{\pm3.88}$ | $-5.73_{\pm7.77}$ | $-3.82_{\pm5.18}$ | $-2.87_{\pm3.90}$ |
| CMAES | $-47.04_{\pm4.38}$ | $0.92_{\pm0.05}$ | $1.85_{\pm0.09}$ | $1.23_{\pm0.06}$ | $1.39_{\pm0.07}$ | - | - | - | - | - | - | - | - | - | - |
| TTDDEA | $-136.49_{\pm46.65}$ | $0.44_{\pm0.28}$ | $0.88_{\pm0.57}$ | $0.58_{\pm0.38}$ | $0.66_{\pm0.43}$ | $-103.75_{\pm20.93}$ | $0.43_{\pm0.24}$ | $0.86_{\pm0.47}$ | $0.57_{\pm0.32}$ | $0.52_{\pm0.28}$ | $-104.54_{\pm27.77}$ | $0.45_{\pm0.09}$ | $0.90_{\pm0.39}$ | $0.60_{\pm0.26}$ | $0.45_{\pm0.20}$ |
| Trimentoring | $-2.79e3_{\pm7.21e3}$ | $-15.83_{\pm44.01}$ | $-31.78_{\pm87.97}$ | $-21.14_{\pm58.66}$ | $-23.84_{\pm66.10}$ | $-2.88e3_{\pm7.44e3}$ | $-18.67_{\pm51.57}$ | $-37.45_{\pm103.10}$ | $-24.93_{\pm68.75}$ | $-22.48_{\pm62.04}$ | $-2.29e3_{\pm5.67e3}$ | $-19.81_{\pm54.61}$ | $-39.74_{\pm109.18}$ | $-26.45_{\pm72.80}$ | $-19.88_{\pm54.80}$ |

Table 14: Overall results for GTOPX unconstrained tasks with 128 solutions and 100th percentile evaluations. In this case, 40% of the values are missing near the worst value and another 10% near the optimal value. Details are the same as Table 5.

| Steps | t = 50 | | | | | t = 100 | | | | | t = 150 | | | | |
|---|---|---|---|---|---|---|---|---|---|---|---|---|---|---|---|
| Task $f(x^*_{\mathrm{OFF}})$ | GTOPX 2 −136.28±0.39 | | | | | | | | | | | | | | |
| Metric | FS | SI | OI | SO | SO$_\omega$ | FS | SI | OI | SO | SO$_\omega$ | FS | SI | OI | SO | SO$_\omega$ |
| ARCOO | −66.14±4.92 | 0.89±0.02 | 1.70±0.08 | 1.17±0.04 | 1.30±0.04 | −64.10±11.63 | 0.89±0.02 | 1.77±0.05 | 1.19±0.03 | 1.07±0.03 | −67.07±11.94 | 0.90±0.02 | 1.80±0.03 | 1.20±0.02 | 0.90±0.02 |
| BO | −86.93±10.93 | 0.75±0.02 | 1.46±0.04 | 0.99±0.02 | 1.11±0.02 | −100.86±11.66 | 0.74±0.02 | 1.48±0.03 | 0.99±0.02 | 0.89±0.02 | −92.23±14.60 | 0.74±0.01 | 1.48±0.03 | 0.99±0.02 | 0.74±0.01 |
| CBAS | −136.28±0.39 | 1.00±0.00 | 1.00±0.00 | 1.00±0.00 | 1.00±0.00 | −136.28±0.39 | 1.00±0.00 | 1.00±0.00 | 1.00±0.00 | 1.00±0.00 | −136.28±0.39 | 1.00±0.00 | 1.00±0.00 | 1.00±0.00 | 1.00±0.00 |
| CCDDEA | −150.40±61.44 | 0.29±0.49 | 0.57±0.08 | 0.38±0.66 | 0.43±0.74 | −1.40e4±3.00e4 | −21.49±47.91 | −42.98±95.82 | −28.65±63.88 | −25.86±57.68 | −6.01e4±1.50e5 | −92.93±223.86 | −185.87±447.73 | −123.91±298.48 | −93.24±224.61 |
| CMAES | −52.79±6.43 | 0.95±0.03 | 1.91±0.06 | 1.27±0.04 | 1.43±0.04 | - | - | - | - | - | - | - | - | - | - |
| TTDDEA | −323.27±66.18 | −0.55±0.47 | −1.10±0.95 | −0.73±0.63 | −0.83±0.71 | −315.61±61.22 | −0.77±0.44 | −1.54±0.87 | −1.03±0.58 | −0.93±0.53 | −317.78±66.76 | −0.84±0.46 | −1.68±0.93 | −1.12±0.62 | −0.84±0.47 |
| Trimentoring | −98.43±38.02 | 0.91±0.15 | 1.44±0.44 | 1.08±0.19 | 1.18±0.23 | −98.56±37.89 | 0.91±0.15 | 1.44±0.44 | 1.09±0.20 | 1.02±0.17 | −98.51±37.94 | 0.91±0.16 | 1.44±0.44 | 1.09±0.20 | 0.91±0.16 |
| Task $f(x^*_{\mathrm{OFF}})$ | GTOPX 3 −102.69±0.49 | | | | | | | | | | | | | | |
| ARCOO | −49.04±4.23 | 0.90±0.03 | 1.76±0.08 | 1.19±0.04 | 1.34±0.05 | −48.82±9.59 | 0.90±0.03 | 1.79±0.06 | 1.20±0.04 | 1.08±0.04 | −50.75±12.10 | 0.89±0.04 | 1.79±0.07 | 1.19±0.05 | 0.90±0.04 |
| BO | −60.00±8.98 | 0.76±0.03 | 1.50±0.04 | 1.01±0.03 | 1.14±0.04 | −69.70±11.06 | 0.75±0.02 | 1.49±0.03 | 1.00±0.02 | 0.90±0.02 | −60.68±10.61 | 0.74±0.01 | 1.49±0.03 | 0.99±0.02 | 0.74±0.01 |
| CBAS | −102.69±0.49 | 1.00±0.00 | 1.00±0.00 | 1.00±0.00 | 1.00±0.00 | −102.69±0.49 | 1.00±0.00 | 1.00±0.00 | 1.00±0.00 | 1.00±0.00 | −102.69±0.49 | 1.00±0.00 | 1.00±0.00 | 1.00±0.00 | 1.00±0.00 |
| CCDDEA | −106.81±46.49 | −11.93±27.75 | −21.69±49.93 | −15.39±35.67 | −17.06±39.48 | −2.51e3±4.37e3 | −9.47±15.92 | −18.93±31.34 | −12.62±21.23 | −11.39±19.16 | −2.57e3±4.34e3 | −12.38±19.37 | −24.75±38.75 | −16.50±25.83 | −12.42±19.44 |
| CMAES | −42.77±5.89 | 0.93±0.02 | 1.87±0.05 | 1.25±0.03 | 1.40±0.04 | - | - | - | - | - | - | - | - | - | - |
| TTDDEA | −134.51±74.25 | 0.31±1.73 | 0.62±1.46 | 0.42±0.98 | 0.47±1.10 | −1.03e5±2.70e5 | −130.55±344.85 | −261.09±689.70 | −174.06±459.80 | −157.07±414.92 | −2.55e5±4.28e5 | −723.43±1638.22 | −1.45e3±3.26e3 | −964.58±2184.29 | −725.84±1643.66 |
| Trimentoring | −88.94±24.03 | 0.93±0.15 | 1.22±0.38 | 1.03±0.17 | 1.08±0.21 | −89.20±23.63 | 0.92±0.16 | 1.21±0.37 | 1.02±0.16 | 0.98±0.15 | −88.08±23.64 | 0.91±0.16 | 1.21±0.36 | 1.01±0.16 | 0.92±0.16 |
| Task $f(x^*_{\mathrm{OFF}})$ | GTOPX 4 −153.62±0.60 | | | | | | | | | | | | | | |
| ARCOO | −84.19±18.96 | 0.87±0.04 | 1.72±0.07 | 1.16±0.04 | 1.30±0.05 | −100.95±24.70 | 0.84±0.03 | 1.68±0.06 | 1.12±0.04 | 1.02±0.04 | −116.14±18.07 | 0.81±0.05 | 1.61±0.10 | 1.08±0.07 | 0.81±0.05 |
| BO | −109.59±12.68 | 0.75±0.03 | 1.46±0.03 | 0.99±0.03 | 1.11±0.03 | −116.90±26.26 | 0.74±0.03 | 1.46±0.03 | 0.98±0.03 | 0.89±0.03 | −108.03±24.17 | 0.73±0.02 | 1.46±0.04 | 0.97±0.02 | 0.73±0.02 |
| CBAS | −153.62±0.60 | 1.00±0.00 | 1.00±0.00 | 1.00±0.00 | 1.00±0.00 | −153.62±0.60 | 1.00±0.00 | 1.00±0.00 | 1.00±0.00 | 1.00±0.00 | −153.62±0.60 | 1.00±0.00 | 1.00±0.00 | 1.00±0.00 | 1.00±0.00 |
| CCDDEA | −275.50±170.25 | −23.54±28.12 | −47.07±56.34 | −31.38±37.49 | −35.36±42.25 | −6.21e5±1.53e6 | −840.61±1.98e3 | −1.68e3±3.96e3 | −1.12e3±2.64e3 | −1.01e3±2.38e3 | −6.29e5±1.53e6 | −1.83e3±4.36e3 | −3.66e3±8.72e3 | −2.44e3±5.83e3 | −1.84e3±4.38e3 |
| CMAES | −98.97±19.36 | 0.83±0.02 | 1.65±0.03 | 1.10±0.02 | 1.24±0.02 | - | - | - | - | - | - | - | - | - | - |
| TTDDEA | −226.38±78.80 | 0.28±0.48 | 0.56±0.95 | 0.37±0.64 | 0.42±0.72 | −273.43±107.88 | 0.02±0.64 | 0.05±1.31 | 0.03±0.87 | 0.03±0.79 | −294.27±115.55 | −0.12±0.72 | −0.24±1.43 | −0.16±0.95 | −0.12±0.72 |
| Trimentoring | −155.72±14.45 | 0.78±0.22 | 1.19±0.33 | 0.92±0.21 | 0.99±0.22 | −147.13±6.86 | 0.77±0.24 | 1.16±0.34 | 0.90±0.23 | 0.85±0.22 | −140.87±17.19 | 0.78±0.22 | 1.19±0.32 | 0.92±0.20 | 0.78±0.22 |
| Task $f(x^*_{\mathrm{OFF}})$ | GTOPX 6 −83.37±0.23 | | | | | | | | | | | | | | |
| ARCOO | −50.45±12.55 | 0.86±0.07 | 1.64±0.15 | 1.13±0.08 | 1.26±0.09 | −50.94±18.33 | 0.84±0.09 | 1.65±0.18 | 1.11±0.12 | 1.00±0.11 | −57.45±20.86 | 0.81±0.10 | 1.63±0.20 | 1.09±0.13 | 0.82±0.10 |
| BO | −58.64±10.23 | 0.75±0.03 | 1.42±0.03 | 0.98±0.03 | 1.10±0.02 | −58.61±11.14 | 0.75±0.03 | 1.42±0.03 | 0.98±0.02 | 0.89±0.02 | −63.46±16.67 | 0.71±0.01 | 1.43±0.02 | 0.95±0.01 | 0.72±0.01 |
| CBAS | −83.37±0.23 | 1.00±0.00 | 1.00±0.00 | 1.00±0.00 | 1.00±0.00 | −83.37±0.23 | 1.00±0.00 | 1.00±0.00 | 1.00±0.00 | 1.00±0.00 | −83.37±0.23 | 1.00±0.00 | 1.00±0.00 | 1.00±0.00 | 1.00±0.00 |
| CCDDEA | −110.12±41.98 | −1.16±2.20 | −2.32±4.40 | −1.54±2.93 | −1.74±3.30 | −7.11e3±1.66e4 | −47.14±91.44 | −94.28±182.88 | −62.86±121.92 | −56.72±110.02 | −7.12e3±1.66e4 | −53.47±87.96 | −106.94±175.92 | −71.29±117.28 | −53.65±88.25 |
| CMAES | −42.11±8.25 | 0.90±0.04 | 1.79±0.09 | 1.19±0.06 | 1.35±0.07 | - | - | - | - | - | - | - | - | - | - |
| TTDDEA | −709.72±1.46e3 | −3.09±14.14 | −6.19±16.29 | −4.12±10.86 | −4.65±12.24 | −834.94±1.62e3 | −4.72±11.92 | −9.45±23.84 | −6.30±15.89 | −5.68±24.34 | −493.95±444.75 | −6.69±15.96 | −13.39±31.92 | −8.92±21.28 | −6.72±16.01 |
| Trimentoring | −62.47±21.34 | 0.89±0.15 | 1.41±0.44 | 1.07±0.18 | 1.16±0.23 | −63.73±20.19 | 0.89±0.15 | 1.41±0.41 | 1.06±0.18 | 1.00±0.15 | −63.24±20.52 | 0.89±0.15 | 1.40±0.40 | 1.06±0.18 | 0.89±0.15 |

Table 15: Overall results for GTOPX unconstrained tasks with 128 solutions and 50th percentile evaluations. In this case, 0% of the values are missing near the worst value and another 50% near the optimal value. Details are the same as Table 5.

| Steps | t = 50 | | | | | t = 100 | | | | | t = 150 | | | | |
|---|---|---|---|---|---|---|---|---|---|---|---|---|---|---|---|
| Task $f(x^*_{\mathrm{OFF}})$ | GTOPX 2 −356.70±2.33 | | | | | | | | | | | | | | |
| Metric | FS | SI | OI | SO | SO$_\omega$ | FS | SI | OI | SO | SO$_\omega$ | FS | SI | OI | SO | SO$_\omega$ |
| ARCOO | −271.14±69.56 | 0.94±0.04 | 1.21±0.17 | 1.05±0.05 | 1.10±0.08 | −273.89±66.42 | 0.96±0.03 | 1.24±0.19 | 1.07±0.06 | 0.97±0.05 | −274.39±65.97 | 0.97±0.03 | 1.25±0.20 | 1.08±0.07 | 0.97±0.05 |
| BO | −382.00±199.67 | 0.65±0.04 | 0.92±0.09 | 0.76±0.06 | 0.81±0.04 | −427.35±75.83 | 0.64±0.03 | 0.93±0.03 | 0.76±0.03 | 0.71±0.03 | −352.27±50.51 | 0.64±0.02 | 0.94±0.02 | 0.76±0.02 | 0.64±0.02 |
| CBAS | −356.70±2.33 | 1.00±0.00 | 1.00±0.00 | 1.00±0.00 | 1.00±0.00 | −356.70±2.33 | 1.00±0.00 | 1.00±0.00 | 1.00±0.00 | 1.00±0.00 | −356.70±2.33 | 1.00±0.00 | 1.00±0.00 | 1.00±0.00 | 1.00±0.00 |
| CCDDEA | −164.79±18.79 | 0.67±0.10 | 1.26±0.21 | 0.88±0.14 | 0.98±0.16 | −2.89e3±3.90e3 | −0.24±1.20 | −0.49±2.31 | −0.33±1.58 | −0.29±1.43 | −3.80e3±4.61e3 | −1.96±5.35 | −3.82±8.53 | −2.59±4.42 | −1.97±3.36 |
| CMAES | −146.67±6.44 | 0.98±0.00 | 1.67±0.02 | 1.24±0.01 | 1.35±0.00 | - | - | - | - | - | - | - | - | - | - |
| TTDDEA | −326.16±271.59 | 0.76±0.35 | 1.26±0.58 | 0.95±0.47 | 1.03±0.47 | −265.98±146.73 | 0.73±0.43 | 1.22±0.70 | 0.91±0.53 | 0.84±0.49 | −261.24±67.42 | 0.74±0.37 | 1.25±0.61 | 0.93±0.46 | 0.75±0.37 |
| Trimentoring | −147.67±7.11 | 0.98±0.01 | 1.67±0.02 | 1.24±0.01 | 1.35±0.01 | −148.51±7.07 | 0.98±0.01 | 1.67±0.02 | 1.24±0.01 | 1.14±0.01 | −149.74±7.48 | 0.98±0.01 | 1.67±0.02 | 1.24±0.01 | 0.99±0.01 |
| Task $f(x^*_{\mathrm{OFF}})$ | GTOPX 3 −336.15±6.28 | | | | | | | | | | | | | | |
| ARCOO | −310.05±54.18 | 0.96±0.04 | 1.08±0.15 | 1.01±0.06 | 1.03±0.08 | −312.22±67.04 | 0.95±0.05 | 1.08±0.18 | 1.00±0.07 | 0.98±0.05 | −313.80±66.60 | 0.94±0.06 | 1.08±0.19 | 1.00±0.08 | 0.94±0.05 |
| BO | −347.83±91.51 | 0.58±0.04 | 0.87±0.07 | 0.70±0.05 | 0.75±0.06 | −313.14±100.73 | 0.57±0.08 | 0.88±0.06 | 0.69±0.05 | 0.65±0.05 | −285.69±46.23 | 0.58±0.02 | 0.89±0.03 | 0.70±0.03 | 0.58±0.03 |
| CBAS | −336.15±6.28 | 1.00±0.00 | 1.00±0.00 | 1.00±0.00 | 1.00±0.00 | −336.15±6.28 | 1.00±0.00 | 1.00±0.00 | 1.00±0.00 | 1.00±0.00 | −336.15±6.28 | 1.00±0.00 | 1.00±0.00 | 1.00±0.00 | 1.00±0.00 |
| CCDDEA | −2.45e4±5.35e4 | −28.84±56.34 | −55.91±112.77 | −38.03±75.14 | −42.62±84.69 | −440.59±437.22 | −18.05±27.69 | −34.88±55.42 | −23.78±36.93 | −21.56±33.32 | −509.41±469.51 | −11.91±16.36 | −23.00±36.76 | −15.68±24.49 | −11.95±18.42 |
| CMAES | −5.81e3±6.00e3 | −1.22±3.44 | −2.02±5.71 | −1.52±4.29 | −1.66±4.68 | - | - | - | - | - | - | - | - | - | - |
| TTDDEA | −3.20e5±5.50e5 | −187.49±377.85 | −200.61±411.58 | −193.77±393.95 | −196.01±399.73 | −1.78e6±4.50e6 | −3.21e3±8.29e3 | −3.50e3±9.05e3 | −3.35e3±8.65e3 | −3.30e3±8.53e3 | −1.98e7±4.68e7 | −8.97e3±1.31e4 | −9.34e3±1.31e4 | −9.15e3±1.31e4 | −5.80e4±3.31e4 |
| Trimentoring | −161.31±70.43 | 0.97±0.01 | 1.57±0.22 | 1.19±0.08 | 1.29±0.12 | −160.47±70.66 | 0.97±0.01 | 1.57±0.22 | 1.20±0.08 | 1.11±0.05 | −159.75±71.07 | 0.97±0.01 | 1.58±0.22 | 1.20±0.08 | 0.98±0.01 |
| Task $f(x^*_{\mathrm{OFF}})$ | GTOPX 4 −496.56±5.01 | | | | | | | | | | | | | | |
| ARCOO | −531.30±221.88 | 0.71±0.35 | 0.95±0.47 | 0.81±0.39 | 0.85±0.41 | −549.33±249.74 | 0.69±0.39 | 0.92±0.52 | 0.78±0.43 | 0.75±0.42 | −536.81±228.96 | 0.67±0.42 | 0.90±0.55 | 0.77±0.47 | 0.68±0.42 |
| BO | −386.97±108.08 | 0.51±0.04 | 0.80±0.05 | 0.62±0.05 | 0.67±0.05 | −575.24±364.21 | 0.53±0.04 | 0.83±0.05 | 0.65±0.04 | 0.60±0.04 | −535.58±165.02 | 0.52±0.03 | 0.83±0.04 | 0.64±0.03 | 0.52±0.03 |
| CBAS | −496.56±5.01 | 1.00±0.00 | 1.00±0.00 | 1.00±0.00 | 1.00±0.00 | −496.56±5.01 | 1.00±0.00 | 1.00±0.00 | 1.00±0.00 | 1.00±0.00 | −496.56±5.01 | 1.00±0.00 | 1.00±0.00 | 1.00±0.00 | 1.00±0.00 |
| CCDDEA | −8.72e3±1.78e4 | −476.38±790.58 | −827.79±1.17e3 | −604.58±1.00e3 | −665.10±1.10e3 | −1.12e7±2.28e7 | −1.56e3±2.53e3 | −2.74e3±4.52e3 | −1.99e3±3.25e3 | −1.83e3±2.97e3 | −2.76e6±6.82e6 | −2.58e3±4.91e3 | −4.52e3±8.76e3 | −3.28e3±6.30e3 | −2.59e3±4.93e3 |
| CMAES | −1.32e4±2.19e4 | −1.05±1.85 | −1.61±2.83 | −1.27±2.34 | −1.37±2.41 | - | - | - | - | - | - | - | - | - | - |
| TTDDEA | −5.10e4±9.15e4 | −3.42e4±9.05e4 | −3.43e4±9.05e4 | −3.42e4±9.05e4 | −3.42e4±9.05e4 | −1.78e7±7.09e4 | −2.69e4±7.09e4 | −2.69e4±7.09e4 | −2.69e4±7.09e4 | −2.69e4±7.09e4 | −9.16e7±2.42e8 | −5.80e4±1.53e5 | −5.80e4±1.53e5 | −5.80e4±1.53e5 | −5.80e4±1.53e5 |
| Trimentoring | −377.02±51.00 | 0.83±0.06 | 1.32±0.13 | 1.02±0.02 | 1.10±0.05 | −377.96±50.06 | 0.82±0.06 | 1.30±0.12 | 1.00±0.01 | 0.93±0.02 | −383.04±48.40 | 0.82±0.06 | 1.29±0.12 | 0.99±0.01 | 0.82±0.06 |
| Task $f(x^*_{\mathrm{OFF}})$ | GTOPX 6 −168.89±0.62 | | | | | | | | | | | | | | |
| ARCOO | −169.34±0.65 | 1.00±0.00 | 1.00±0.00 | 1.00±0.00 | 1.00±0.00 | −169.51±0.66 | 1.00±0.00 | 1.00±0.00 | 1.00±0.00 | 1.00±0.00 | −169.54±0.66 | 1.00±0.00 | 1.00±0.00 | 1.00±0.00 | 1.00±0.00 |
| BO | −165.55±21.61 | 0.74±0.05 | 0.97±0.03 | 0.84±0.04 | 0.88±0.02 | −178.18±38.05 | 0.71±0.05 | 0.97±0.01 | 0.82±0.02 | 0.78±0.02 | −176.87±38.63 | 0.70±0.04 | 0.97±0.06 | 0.81±0.03 | 0.70±0.02 |
| CBAS | −168.89±0.62 | 1.00±0.00 | 1.00±0.00 | 1.00±0.00 | 1.00±0.00 | −168.89±0.62 | 1.00±0.00 | 1.00±0.00 | 1.00±0.00 | 1.00±0.00 | −168.89±0.62 | 1.00±0.00 | 1.00±0.00 | 1.00±0.00 | 1.00±0.00 |
| CCDDEA | −190.50±114.32 | 0.46±0.34 | 0.84±0.63 | 0.59±0.44 | 0.66±0.49 | −579.63±480.01 | −0.02±0.73 | 0.05±1.24 | 0.00±0.92 | −0.01±0.85 | −579.63±480.00 | −0.49±1.27 | −0.75±2.11 | −0.60±1.39 | −0.49±1.28 |
| CMAES | −146.55±5.52 | 0.93±0.02 | 1.17±0.03 | 1.03±0.02 | 1.08±0.02 | - | - | - | - | - | - | - | - | - | - |
| TTDDEA | −625.37±927.64 | −0.97±3.39 | −1.22±4.76 | −1.09±3.96 | −1.14±4.20 | −666.70±1.05e3 | −1.48±3.94 | −1.91±5.62 | −1.68±4.63 | −1.61±4.38 | −3.95e3±9.83e3 | −3.91±9.60 | −5.53±14.13 | −4.59±11.43 | −3.92±9.62 |
| Trimentoring | −155.86±5.60 | −12.85±36.16 | −12.57±36.28 | −12.73±36.20 | −12.68±36.22 | −157.22±4.61 | −12.93±36.34 | −12.66±36.45 | −12.81±36.39 | −12.86±36.57 | −158.01±4.65 | −12.96±36.40 | −12.69±36.51 | −12.84±36.45 | −12.95±36.40 |

Table 16: Overall results for GTOPX unconstrained tasks with 128 solutions and 50th percentile evaluations. In this case, 0% of the values are missing near the worst value and another 40% near the optimal value. Details are the same as Table 5.

| Steps | t = 50 | | | | | t = 100 | | | | | t = 150 | | | | |
|---|---|---|---|---|---|---|---|---|---|---|---|---|---|---|---|
| **Task** $f(x^*_{OFF})$ | GTOPX 2 $-275.27_{\pm2.34}$ | | | | | | | | | | | | | | |
| Metric | FS | SI | OI | SO | SO$_w$ | FS | SI | OI | SO | SO$_w$ | FS | SI | OI | SO | SO$_w$ |
| ARCOO | $-240.43_{\pm29.36}$ | $0.95_{\pm0.03}$ | $1.12_{\pm0.11}$ | $1.02_{\pm0.04}$ | $1.05_{\pm0.06}$ | $-240.03_{\pm28.21}$ | $0.96_{\pm0.03}$ | $1.14_{\pm0.12}$ | $1.04_{\pm0.05}$ | $1.01_{\pm0.03}$ | $-240.92_{\pm27.51}$ | $0.96_{\pm0.02}$ | $1.15_{\pm0.12}$ | $1.04_{\pm0.05}$ | $0.96_{\pm0.02}$ |
| BO | $-398.56_{\pm136.59}$ | $0.47_{\pm0.03}$ | $0.58_{\pm0.05}$ | $0.52_{\pm0.04}$ | $0.54_{\pm0.04}$ | $-398.53_{\pm78.74}$ | $0.45_{\pm0.03}$ | $0.58_{\pm0.03}$ | $0.51_{\pm0.03}$ | $0.49_{\pm0.03}$ | $-421.96_{\pm93.49}$ | $0.44_{\pm0.04}$ | $0.56_{\pm0.04}$ | $0.49_{\pm0.04}$ | $0.44_{\pm0.04}$ |
| CBAS | $-275.27_{\pm2.34}$ | $1.00_{\pm0.00}$ | $1.00_{\pm0.00}$ | $1.00_{\pm0.00}$ | $1.00_{\pm0.00}$ | $-275.27_{\pm2.34}$ | $1.00_{\pm0.00}$ | $1.00_{\pm0.00}$ | $1.00_{\pm0.00}$ | $1.00_{\pm0.00}$ | $-275.27_{\pm2.34}$ | $1.00_{\pm0.00}$ | $1.00_{\pm0.00}$ | $1.00_{\pm0.00}$ | $1.00_{\pm0.00}$ |
| CCDDEA | $-273.25_{\pm170.45}$ | $0.55_{\pm0.21}$ | $0.99_{\pm0.39}$ | $0.70_{\pm0.26}$ | $0.78_{\pm0.30}$ | $-1.41e3_{\pm1.84e3}$ | $-1.46_{\pm4.34}$ | $-2.67_{\pm7.98}$ | $-1.88_{\pm5.63}$ | $-1.72_{\pm5.13}$ | $-3.93e3_{\pm8.54e3}$ | $-3.56_{\pm9.25}$ | $-6.50_{\pm17.01}$ | $-4.60_{\pm11.99}$ | $-3.57_{\pm9.28}$ |
| CMAES | $-145.33_{\pm7.48}$ | $0.97_{\pm0.01}$ | $1.56_{\pm0.02}$ | $1.20_{\pm0.01}$ | $1.30_{\pm0.00}$ | - | - | - | - | - | - | - | - | - | - |
| TTDDEA | $-663.67_{\pm994.32}$ | $0.43_{\pm0.77}$ | $0.77_{\pm1.04}$ | $0.56_{\pm0.88}$ | $0.62_{\pm0.93}$ | $-988.93_{\pm1.63e3}$ | $-0.47_{\pm2.89}$ | $-0.21_{\pm3.23}$ | $-0.38_{\pm3.03}$ | $-0.41_{\pm2.98}$ | $-1.07e3_{\pm1.74e3}$ | $-1.57_{\pm5.38}$ | $-1.41_{\pm5.87}$ | $-1.52_{\pm5.39}$ | $-1.57_{\pm5.38}$ |
| Trimentoring | $-164.59_{\pm42.36}$ | $0.98_{\pm0.02}$ | $1.49_{\pm0.19}$ | $1.17_{\pm0.07}$ | $1.26_{\pm0.10}$ | $-168.04_{\pm40.94}$ | $0.97_{\pm0.02}$ | $1.49_{\pm0.19}$ | $1.17_{\pm0.07}$ | $1.10_{\pm0.04}$ | $-167.14_{\pm41.18}$ | $0.97_{\pm0.02}$ | $1.49_{\pm0.19}$ | $1.17_{\pm0.07}$ | $0.98_{\pm0.02}$ |
| **Task** $f(x^*_{OFF})$ | GTOPX 3 $-228.84_{\pm3.46}$ | | | | | | | | | | | | | | |
| ARCOO | $-217.56_{\pm12.50}$ | $0.95_{\pm0.03}$ | $1.06_{\pm0.08}$ | $1.00_{\pm0.02}$ | $1.02_{\pm0.03}$ | $-217.44_{\pm18.52}$ | $0.95_{\pm0.04}$ | $1.06_{\pm0.08}$ | $1.00_{\pm0.02}$ | $0.98_{\pm0.01}$ | $-219.50_{\pm16.41}$ | $0.95_{\pm0.04}$ | $1.06_{\pm0.09}$ | $1.00_{\pm0.03}$ | $0.95_{\pm0.04}$ |
| BO | $-517.73_{\pm213.88}$ | $0.21_{\pm0.04}$ | $0.25_{\pm0.07}$ | $0.23_{\pm0.05}$ | $0.24_{\pm0.06}$ | $-381.92_{\pm151.11}$ | $0.18_{\pm0.06}$ | $0.23_{\pm0.07}$ | $0.20_{\pm0.06}$ | $0.19_{\pm0.06}$ | $-427.79_{\pm97.84}$ | $0.17_{\pm0.04}$ | $0.22_{\pm0.05}$ | $0.19_{\pm0.04}$ | $0.17_{\pm0.04}$ |
| CBAS | $-228.84_{\pm3.46}$ | $1.00_{\pm0.00}$ | $1.00_{\pm0.00}$ | $1.00_{\pm0.00}$ | $1.00_{\pm0.00}$ | $-228.84_{\pm3.46}$ | $1.00_{\pm0.00}$ | $1.00_{\pm0.00}$ | $1.00_{\pm0.00}$ | $1.00_{\pm0.00}$ | $-228.84_{\pm3.46}$ | $1.00_{\pm0.00}$ | $1.00_{\pm0.00}$ | $1.00_{\pm0.00}$ | $1.00_{\pm0.00}$ |
| CCDDEA | $-1.66e3_{\pm2.26e3}$ | $-3.02_{\pm4.41}$ | $-5.50_{\pm8.16}$ | $-3.90_{\pm5.73}$ | $-4.32_{\pm6.37}$ | $-294.20_{\pm339.28}$ | $-6.62_{\pm12.39}$ | $-12.46_{\pm23.55}$ | $-8.65_{\pm16.24}$ | $-7.87_{\pm14.76}$ | $-294.16_{\pm339.29}$ | $-4.28_{\pm8.55}$ | $-8.06_{\pm16.24}$ | $-5.59_{\pm11.20}$ | $-4.29_{\pm8.58}$ |
| CMAES | $-4.75e3_{\pm4.14e3}$ | $-1.14_{\pm3.98}$ | $-1.70_{\pm4.00}$ | $-1.36_{\pm4.79}$ | $-1.46_{\pm5.13}$ | - | - | - | - | - | - | - | - | - | - |
| TTDDEA | $-8.29e6_{\pm1.39e7}$ | $-1.02e4_{\pm2.37e4}$ | $-1.02e4_{\pm2.37e4}$ | $-1.02e4_{\pm2.37e4}$ | $-1.02e4_{\pm2.37e4}$ | $-4.50e6_{\pm1.10e7}$ | $-1.61e4_{\pm3.67e4}$ | $-1.62e4_{\pm3.67e4}$ | $-1.61e4_{\pm3.67e4}$ | $-1.61e4_{\pm3.67e4}$ | $-3.08e6_{\pm4.94e6}$ | $-6.73e4_{\pm1.35e5}$ | $-6.73e4_{\pm1.35e5}$ | $-6.73e4_{\pm1.35e5}$ | $-6.73e4_{\pm1.35e5}$ |
| Trimentoring | $-140.22_{\pm19.44}$ | $0.95_{\pm0.03}$ | $1.44_{\pm0.08}$ | $1.15_{\pm0.04}$ | $1.23_{\pm0.05}$ | $-140.01_{\pm14.02}$ | $0.96_{\pm0.02}$ | $1.45_{\pm0.08}$ | $1.15_{\pm0.04}$ | $1.08_{\pm0.03}$ | $-141.41_{\pm13.23}$ | $0.96_{\pm0.02}$ | $1.45_{\pm0.07}$ | $1.16_{\pm0.04}$ | $0.96_{\pm0.02}$ |
| **Task** $f(x^*_{OFF})$ | GTOPX 4 $-322.50_{\pm1.08}$ | | | | | | | | | | | | | | |
| ARCOO | $-383.93_{\pm142.87}$ | $0.74_{\pm0.46}$ | $0.80_{\pm0.48}$ | $0.77_{\pm0.47}$ | $0.78_{\pm0.47}$ | $-404.32_{\pm204.47}$ | $0.66_{\pm0.65}$ | $0.72_{\pm0.67}$ | $0.69_{\pm0.66}$ | $0.68_{\pm0.66}$ | $-392.43_{\pm172.40}$ | $0.64_{\pm0.72}$ | $0.70_{\pm0.74}$ | $0.67_{\pm0.72}$ | $0.64_{\pm0.72}$ |
| BO | $-441.36_{\pm145.26}$ | $0.01_{\pm0.09}$ | $0.01_{\pm0.11}$ | $0.01_{\pm0.10}$ | $0.01_{\pm0.10}$ | $-541.09_{\pm231.09}$ | $0.00_{\pm0.08}$ | $0.00_{\pm0.10}$ | $0.00_{\pm0.09}$ | $0.00_{\pm0.09}$ | $-437.02_{\pm102.33}$ | $-0.02_{\pm0.07}$ | $-0.02_{\pm0.09}$ | $-0.02_{\pm0.08}$ | $-0.02_{\pm0.07}$ |
| CBAS | $-322.50_{\pm1.08}$ | $1.00_{\pm0.00}$ | $1.00_{\pm0.00}$ | $1.00_{\pm0.00}$ | $1.00_{\pm0.00}$ | $-322.50_{\pm1.08}$ | $1.00_{\pm0.00}$ | $1.00_{\pm0.00}$ | $1.00_{\pm0.00}$ | $1.00_{\pm0.00}$ | $-322.50_{\pm1.08}$ | $1.00_{\pm0.00}$ | $1.00_{\pm0.00}$ | $1.00_{\pm0.00}$ | $1.00_{\pm0.00}$ |
| CCDDEA | $-1.09e5_{\pm2.15e5}$ | $-369.06_{\pm599.13}$ | $-594.34_{\pm969.19}$ | $-455.41_{\pm739.53}$ | $-494.30_{\pm803.30}$ | $-7.72e6_{\pm1.99e7}$ | $-5.85e3_{\pm9.66e3}$ | $-9.77e3_{\pm1.64e4}$ | $-7.31e3_{\pm1.22e4}$ | $-6.76e3_{\pm1.12e4}$ | $-3.41e5_{\pm4.06e5}$ | $-4.92e3_{\pm7.45e3}$ | $-8.18e3_{\pm1.25e4}$ | $-6.15e3_{\pm9.35e3}$ | $-4.94e3_{\pm7.47e3}$ |
| CMAES | $-9.15e3_{\pm1.28e4}$ | $-1.92_{\pm3.99}$ | $-2.35_{\pm4.90}$ | $-2.11_{\pm4.39}$ | $-2.19_{\pm4.55}$ | - | - | - | - | - | - | - | - | - | - |
| TTDDEA | $-1.22e4_{\pm2.84e4}$ | $-72.95_{\pm179.32}$ | $-72.57_{\pm179.48}$ | $-72.81_{\pm179.38}$ | $-72.74_{\pm179.41}$ | $-7.64e6_{\pm2.02e7}$ | $-1.06e3_{\pm2.79e3}$ | $-1.06e3_{\pm2.79e3}$ | $-1.06e3_{\pm2.79e3}$ | $-1.06e3_{\pm2.79e3}$ | $-6.65e4_{\pm1.58e5}$ | $-6.67e3_{\pm1.76e4}$ | $-6.67e3_{\pm1.76e4}$ | $-6.67e3_{\pm1.76e4}$ | $-6.67e3_{\pm1.76e4}$ |
| Trimentoring | $-541.87_{\pm474.18}$ | $0.16_{\pm1.24}$ | $0.28_{\pm1.66}$ | $0.21_{\pm1.42}$ | $0.23_{\pm1.49}$ | $-542.42_{\pm464.75}$ | $0.11_{\pm1.32}$ | $0.20_{\pm1.76}$ | $0.14_{\pm1.51}$ | $0.13_{\pm1.44}$ | $-557.83_{\pm502.94}$ | $0.08_{\pm1.37}$ | $0.16_{\pm1.83}$ | $0.11_{\pm1.56}$ | $0.08_{\pm1.37}$ |
| **Task** $f(x^*_{OFF})$ | GTOPX 6 $-142.09_{\pm0.39}$ | | | | | | | | | | | | | | |
| ARCOO | $-142.75_{\pm0.48}$ | $0.99_{\pm0.00}$ | $0.99_{\pm0.00}$ | $0.99_{\pm0.00}$ | $0.99_{\pm0.00}$ | $-142.88_{\pm0.52}$ | $0.99_{\pm0.00}$ | $0.99_{\pm0.00}$ | $0.99_{\pm0.00}$ | $0.99_{\pm0.00}$ | $-142.90_{\pm0.51}$ | $0.99_{\pm0.00}$ | $0.99_{\pm0.00}$ | $0.99_{\pm0.00}$ | $0.99_{\pm0.00}$ |
| BO | $-161.78_{\pm28.78}$ | $0.63_{\pm0.05}$ | $0.72_{\pm0.02}$ | $0.67_{\pm0.04}$ | $0.69_{\pm0.03}$ | $-189.56_{\pm52.36}$ | $0.59_{\pm0.01}$ | $0.71_{\pm0.02}$ | $0.65_{\pm0.02}$ | $0.63_{\pm0.03}$ | $-167.35_{\pm17.92}$ | $0.59_{\pm0.03}$ | $0.72_{\pm0.02}$ | $0.65_{\pm0.02}$ | $0.59_{\pm0.03}$ |
| CBAS | $-142.09_{\pm0.39}$ | $1.00_{\pm0.00}$ | $1.00_{\pm0.00}$ | $1.00_{\pm0.00}$ | $1.00_{\pm0.00}$ | $-142.09_{\pm0.39}$ | $1.00_{\pm0.00}$ | $1.00_{\pm0.00}$ | $1.00_{\pm0.00}$ | $1.00_{\pm0.00}$ | $-142.09_{\pm0.39}$ | $1.00_{\pm0.00}$ | $1.00_{\pm0.00}$ | $1.00_{\pm0.00}$ | $1.00_{\pm0.00}$ |
| CCDDEA | $-212.20_{\pm122.84}$ | $0.03_{\pm0.41}$ | $0.04_{\pm0.73}$ | $0.03_{\pm0.52}$ | $0.04_{\pm0.56}$ | $-257.12_{\pm191.11}$ | $-0.14_{\pm0.70}$ | $-0.19_{\pm1.15}$ | $-0.16_{\pm0.87}$ | $-0.15_{\pm0.81}$ | $-198.51_{\pm107.90}$ | $-0.07_{\pm0.64}$ | $-0.08_{\pm1.09}$ | $-0.07_{\pm0.81}$ | $-0.07_{\pm0.65}$ |
| CMAES | $-152.86_{\pm5.17}$ | $0.88_{\pm0.03}$ | $0.92_{\pm0.04}$ | $0.90_{\pm0.03}$ | $0.91_{\pm0.03}$ | - | - | - | - | - | - | - | - | - | - |
| TTDDEA | $-119.09_{\pm33.02}$ | $0.78_{\pm0.11}$ | $1.20_{\pm0.21}$ | $0.94_{\pm0.14}$ | $1.02_{\pm0.16}$ | $-123.76_{\pm38.40}$ | $0.78_{\pm0.11}$ | $1.22_{\pm0.23}$ | $0.95_{\pm0.15}$ | $0.89_{\pm0.13}$ | $-132.60_{\pm38.43}$ | $0.76_{\pm0.11}$ | $1.20_{\pm0.20}$ | $0.93_{\pm0.16}$ | $0.76_{\pm0.11}$ |
| Trimentoring | $-156.94_{\pm7.90}$ | $-7.63_{\pm22.05}$ | $-7.48_{\pm22.11}$ | $-7.56_{\pm22.08}$ | $-7.54_{\pm22.09}$ | $-157.48_{\pm8.01}$ | $-7.69_{\pm22.16}$ | $-7.54_{\pm22.22}$ | $-7.62_{\pm22.19}$ | $-7.64_{\pm22.18}$ | $-157.08_{\pm7.09}$ | $-7.71_{\pm22.30}$ | $-7.56_{\pm22.25}$ | $-7.64_{\pm22.22}$ | $-7.71_{\pm22.30}$ |

Table 17: Overall results for GTOPX unconstrained tasks with 128 solutions and 50th percentile evaluations. In this case, 0% of the values are missing near the worst value and another 30% near the optimal value. Details are the same as Table 5.

| Steps | t = 50 | | | | | t = 100 | | | | | t = 150 | | | | |
|---|---|---|---|---|---|---|---|---|---|---|---|---|---|---|---|
| **Task** $f(x^*_{OFF})$ | GTOPX 2 $-218.12_{\pm1.78}$ | | | | | | | | | | | | | | |
| Metric | FS | SI | OI | SO | SO$_w$ | FS | SI | OI | SO | SO$_w$ | FS | SI | OI | SO | SO$_w$ |
| ARCOO | $-211.50_{\pm20.06}$ | $0.96_{\pm0.03}$ | $1.04_{\pm0.06}$ | $1.00_{\pm0.03}$ | $1.01_{\pm0.03}$ | $-212.34_{\pm9.57}$ | $0.96_{\pm0.03}$ | $1.04_{\pm0.05}$ | $1.00_{\pm0.03}$ | $0.99_{\pm0.02}$ | $-211.26_{\pm9.82}$ | $0.96_{\pm0.01}$ | $1.04_{\pm0.06}$ | $1.00_{\pm0.03}$ | $0.96_{\pm0.03}$ |
| BO | $-367.62_{\pm98.35}$ | $0.05_{\pm0.10}$ | $0.05_{\pm0.11}$ | $0.05_{\pm0.10}$ | $0.05_{\pm0.10}$ | $-333.32_{\pm59.64}$ | $0.05_{\pm0.08}$ | $0.05_{\pm0.08}$ | $0.05_{\pm0.08}$ | $0.05_{\pm0.08}$ | $-360.47_{\pm92.18}$ | $0.04_{\pm0.06}$ | $0.04_{\pm0.07}$ | $0.04_{\pm0.06}$ | $0.04_{\pm0.06}$ |
| CBAS | $-218.12_{\pm1.78}$ | $1.00_{\pm0.00}$ | $1.00_{\pm0.00}$ | $1.00_{\pm0.00}$ | $1.00_{\pm0.00}$ | $-218.12_{\pm1.78}$ | $1.00_{\pm0.00}$ | $1.00_{\pm0.00}$ | $1.00_{\pm0.00}$ | $1.00_{\pm0.00}$ | $-218.12_{\pm1.78}$ | $1.00_{\pm0.00}$ | $1.00_{\pm0.00}$ | $1.00_{\pm0.00}$ | $1.00_{\pm0.00}$ |
| CCDDEA | $-299.19_{\pm279.77}$ | $0.10_{\pm0.65}$ | $0.15_{\pm1.24}$ | $0.12_{\pm0.85}$ | $0.13_{\pm0.95}$ | $-3.92e3_{\pm4.42e3}$ | $-3.87_{\pm4.15}$ | $-7.12_{\pm7.94}$ | $-5.01_{\pm5.44}$ | $-4.57_{\pm4.94}$ | $-7.55e3_{\pm1.16e4}$ | $-11.03_{\pm18.83}$ | $-20.95_{\pm36.64}$ | $-14.44_{\pm24.88}$ | $-11.06_{\pm18.89}$ |
| CMAES | $-146.45_{\pm3.49}$ | $0.96_{\pm0.01}$ | $1.42_{\pm0.02}$ | $1.14_{\pm0.01}$ | $1.22_{\pm0.01}$ | - | - | - | - | - | - | - | - | - | - |
| TTDDEA | $-272.39_{\pm122.30}$ | $0.65_{\pm0.28}$ | $0.84_{\pm0.36}$ | $0.73_{\pm0.31}$ | $0.77_{\pm0.33}$ | $-268.24_{\pm93.92}$ | $0.59_{\pm0.38}$ | $0.77_{\pm0.50}$ | $0.67_{\pm0.43}$ | $0.64_{\pm0.41}$ | $-256.68_{\pm71.26}$ | $0.57_{\pm0.39}$ | $0.75_{\pm0.50}$ | $0.65_{\pm0.43}$ | $0.58_{\pm0.39}$ |
| Trimentoring | $-173.94_{\pm34.69}$ | $-14.90_{\pm41.94}$ | $-14.59_{\pm42.05}$ | $-14.78_{\pm41.99}$ | $-14.72_{\pm42.01}$ | $-175.43_{\pm34.01}$ | $-14.98_{\pm42.16}$ | $-14.68_{\pm42.27}$ | $-14.86_{\pm42.20}$ | $-14.91_{\pm42.19}$ | $-175.87_{\pm33.85}$ | $-15.01_{\pm42.23}$ | $-14.71_{\pm42.34}$ | $-14.89_{\pm42.27}$ | $-15.01_{\pm42.23}$ |
| **Task** $f(x^*_{OFF})$ | GTOPX 3 $-171.54_{\pm0.81}$ | | | | | | | | | | | | | | |
| ARCOO | $-172.31_{\pm0.70}$ | $0.99_{\pm0.02}$ | $1.00_{\pm0.00}$ | $0.99_{\pm0.01}$ | $0.99_{\pm0.01}$ | $-172.10_{\pm0.69}$ | $0.99_{\pm0.02}$ | $1.00_{\pm0.00}$ | $0.99_{\pm0.01}$ | $0.99_{\pm0.01}$ | $-172.02_{\pm0.74}$ | $0.98_{\pm0.02}$ | $1.00_{\pm0.00}$ | $0.99_{\pm0.01}$ | $0.98_{\pm0.02}$ |
| BO | $-377.49_{\pm144.48}$ | $-0.40_{\pm0.11}$ | $-0.40_{\pm0.11}$ | $-0.40_{\pm0.11}$ | $-0.40_{\pm0.11}$ | $-574.87_{\pm288.57}$ | $-0.45_{\pm0.06}$ | $-0.46_{\pm0.07}$ | $-0.46_{\pm0.07}$ | $-0.45_{\pm0.07}$ | $-508.79_{\pm263.53}$ | $-0.46_{\pm0.06}$ | $-0.48_{\pm0.08}$ | $-0.47_{\pm0.06}$ | $-0.46_{\pm0.06}$ |
| CBAS | $-171.54_{\pm0.81}$ | $1.00_{\pm0.00}$ | $1.00_{\pm0.00}$ | $1.00_{\pm0.00}$ | $1.00_{\pm0.00}$ | $-171.54_{\pm0.81}$ | $1.00_{\pm0.00}$ | $1.00_{\pm0.00}$ | $1.00_{\pm0.00}$ | $1.00_{\pm0.00}$ | $-171.54_{\pm0.81}$ | $1.00_{\pm0.00}$ | $1.00_{\pm0.00}$ | $1.00_{\pm0.00}$ | $1.00_{\pm0.00}$ |
| CCDDEA | $-3.94e3_{\pm5.00e3}$ | $-27.99_{\pm39.23}$ | $-44.35_{\pm56.45}$ | $-34.10_{\pm42.93}$ | $-36.91_{\pm46.57}$ | $-690.61_{\pm1.07e3}$ | $-20.39_{\pm19.65}$ | $-34.87_{\pm34.22}$ | $-25.55_{\pm24.79}$ | $-23.64_{\pm22.82}$ | $-733.08_{\pm1.08e3}$ | $-14.14_{\pm12.79}$ | $-24.26_{\pm22.27}$ | $-17.79_{\pm16.14}$ | $-14.18_{\pm12.82}$ |
| CMAES | $-3.19e3_{\pm4.46e3}$ | $-3.04_{\pm9.23}$ | $-3.86_{\pm11.76}$ | $-3.40_{\pm10.34}$ | $-3.54_{\pm10.78}$ | - | - | - | - | - | - | - | - | - | - |
| TTDDEA | $-8.55e6_{\pm2.14e7}$ | $-1.18e4_{\pm2.64e4}$ | $-1.18e4_{\pm2.64e4}$ | $-1.18e4_{\pm2.64e4}$ | $-1.18e4_{\pm2.64e4}$ | $-8.96e7_{\pm2.35e8}$ | $-6.95e4_{\pm1.76e5}$ | $-6.95e4_{\pm1.76e5}$ | $-6.95e4_{\pm1.76e5}$ | $-6.95e4_{\pm1.76e5}$ | $-1.13e8_{\pm2.89e8}$ | $-4.87e5_{\pm1.18e6}$ | $-4.88e5_{\pm1.18e6}$ | $-4.87e5_{\pm1.18e6}$ | $-4.87e5_{\pm1.18e6}$ |
| Trimentoring | $-136.90_{\pm6.14}$ | $0.92_{\pm0.04}$ | $1.24_{\pm0.05}$ | $1.06_{\pm0.02}$ | $1.11_{\pm0.02}$ | $-136.26_{\pm5.73}$ | $0.92_{\pm0.03}$ | $1.25_{\pm0.03}$ | $1.06_{\pm0.03}$ | $1.01_{\pm0.03}$ | $-132.64_{\pm7.71}$ | $0.93_{\pm0.03}$ | $1.25_{\pm0.04}$ | $1.06_{\pm0.03}$ | $0.93_{\pm0.03}$ |
| **Task** $f(x^*_{OFF})$ | GTOPX 4 $-241.61_{\pm1.39}$ | | | | | | | | | | | | | | |
| ARCOO | $-415.49_{\pm206.56}$ | $0.24_{\pm0.78}$ | $0.25_{\pm0.78}$ | $0.24_{\pm0.78}$ | $0.24_{\pm0.78}$ | $-427.11_{\pm270.86}$ | $0.07_{\pm1.12}$ | $0.08_{\pm1.12}$ | $0.07_{\pm1.12}$ | $0.07_{\pm1.12}$ | $-415.59_{\pm237.43}$ | $0.02_{\pm1.24}$ | $0.03_{\pm1.24}$ | $0.03_{\pm1.24}$ | $0.02_{\pm1.24}$ |
| BO | $-780.90_{\pm389.21}$ | $-1.00_{\pm0.17}$ | $-1.01_{\pm0.19}$ | $-1.00_{\pm0.18}$ | $-1.00_{\pm0.18}$ | $-591.69_{\pm326.36}$ | $-0.94_{\pm0.15}$ | $-0.95_{\pm0.18}$ | $-0.94_{\pm0.15}$ | $-0.94_{\pm0.15}$ | $-787.97_{\pm357.01}$ | $-0.89_{\pm0.11}$ | $-0.91_{\pm0.11}$ | $-0.90_{\pm0.11}$ | $-0.89_{\pm0.11}$ |
| CBAS | $-241.61_{\pm1.39}$ | $1.00_{\pm0.00}$ | $1.00_{\pm0.00}$ | $1.00_{\pm0.00}$ | $1.00_{\pm0.00}$ | $-241.61_{\pm1.39}$ | $1.00_{\pm0.00}$ | $1.00_{\pm0.00}$ | $1.00_{\pm0.00}$ | $1.00_{\pm0.00}$ | $-241.61_{\pm1.39}$ | $1.00_{\pm0.00}$ | $1.00_{\pm0.00}$ | $1.00_{\pm0.00}$ | $1.00_{\pm0.00}$ |
| CCDDEA | $-3.36e3_{\pm2.52e5}$ | $-7.27e3_{\pm1.71e4}$ | $-9.64e3_{\pm2.30e4}$ | $-8.28e3_{\pm1.99e4}$ | $-8.69e3_{\pm2.09e4}$ | $-2.46e5_{\pm2.42e5}$ | $-3.37e4_{\pm7.74e4}$ | $-5.85e4_{\pm1.39e5}$ | $-4.27e4_{\pm9.84e4}$ | $-3.93e4_{\pm9.10e4}$ | $-5.79e5_{\pm9.88e5}$ | $-5.10e4_{\pm1.23e5}$ | $-8.95e4_{\pm2.19e5}$ | $-6.49e4_{\pm1.57e5}$ | $-5.12e4_{\pm1.23e5}$ |
| CMAES | $-8.92e3_{\pm2.24e4}$ | $-5.63_{\pm16.34}$ | $-5.63_{\pm16.34}$ | $-5.63_{\pm16.34}$ | $-5.63_{\pm16.34}$ | - | - | - | - | - | - | - | - | - | - |
| TTDDEA | $-267.21_{\pm108.42}$ | $0.48_{\pm0.52}$ | $0.71_{\pm0.50}$ | $0.57_{\pm0.39}$ | $0.61_{\pm0.42}$ | $-1.08e3_{\pm1.98e3}$ | $-0.30_{\pm2.10}$ | $-0.42_{\pm3.05}$ | $-0.35_{\pm2.49}$ | $-0.34_{\pm2.35}$ | $-1.37e3_{\pm2.85e3}$ | $-1.91_{\pm6.10}$ | $-2.72_{\pm8.93}$ | $-2.24_{\pm7.25}$ | $-1.91_{\pm6.12}$ |
| Trimentoring | $-501.55_{\pm415.22}$ | $-0.31_{\pm1.98}$ | $-0.34_{\pm2.23}$ | $-0.33_{\pm2.10}$ | $-0.33_{\pm2.14}$ | $-505.77_{\pm404.23}$ | $-0.43_{\pm2.15}$ | $-0.47_{\pm2.42}$ | $-0.45_{\pm2.28}$ | $-0.44_{\pm2.34}$ | $-521.76_{\pm441.22}$ | $-0.47_{\pm2.22}$ | $-0.52_{\pm2.50}$ | $-0.50_{\pm2.35}$ | $-0.47_{\pm2.22}$ |
| **Task** $f(x^*_{OFF})$ | GTOPX 6 $-121.48_{\pm0.38}$ | | | | | | | | | | | | | | |
| ARCOO | $-122.23_{\pm0.43}$ | $0.99_{\pm0.00}$ | $0.99_{\pm0.00}$ | $0.99_{\pm0.00}$ | $0.99_{\pm0.00}$ | $-122.30_{\pm0.42}$ | $0.99_{\pm0.00}$ | $0.99_{\pm0.00}$ | $0.99_{\pm0.00}$ | $0.99_{\pm0.00}$ | $-122.34_{\pm0.41}$ | $0.99_{\pm0.00}$ | $0.99_{\pm0.00}$ | $0.99_{\pm0.00}$ | $0.99_{\pm0.00}$ |
| BO | $-182.51_{\pm31.27}$ | $0.42_{\pm0.08}$ | $0.43_{\pm0.04}$ | $0.43_{\pm0.04}$ | $0.43_{\pm0.04}$ | $-174.67_{\pm16.85}$ | $0.41_{\pm0.03}$ | $0.42_{\pm0.03}$ | $0.42_{\pm0.04}$ | $0.41_{\pm0.03}$ | $-183.31_{\pm20.79}$ | $0.41_{\pm0.02}$ | $0.42_{\pm0.04}$ | $0.41_{\pm0.03}$ | $0.41_{\pm0.03}$ |
| CBAS | $-121.48_{\pm0.38}$ | $1.00_{\pm0.00}$ | $1.00_{\pm0.00}$ | $1.00_{\pm0.00}$ | $1.00_{\pm0.00}$ | $-121.48_{\pm0.38}$ | $1.00_{\pm0.00}$ | $1.00_{\pm0.00}$ | $1.00_{\pm0.00}$ | $1.00_{\pm0.00}$ | $-121.48_{\pm0.38}$ | $1.00_{\pm0.00}$ | $1.00_{\pm0.00}$ | $1.00_{\pm0.00}$ | $1.00_{\pm0.00}$ |
| CCDDEA | $-132.96_{\pm36.93}$ | $-0.09_{\pm0.70}$ | $-0.12_{\pm1.25}$ | $-0.10_{\pm0.97}$ | $-0.11_{\pm1.05}$ | $-207.18_{\pm120.25}$ | $0.08_{\pm0.40}$ | $0.14_{\pm0.74}$ | $0.10_{\pm0.52}$ | $0.09_{\pm0.47}$ | $-208.17_{\pm119.36}$ | $0.04_{\pm0.44}$ | $0.07_{\pm0.81}$ | $0.05_{\pm0.57}$ | $0.04_{\pm0.44}$ |
| CMAES | $-151.03_{\pm4.06}$ | $0.68_{\pm0.07}$ | $0.68_{\pm0.07}$ | $0.68_{\pm0.07}$ | $0.68_{\pm0.07}$ | - | - | - | - | - | - | - | - | - | - |
| TTDDEA | $-263.52_{\pm299.86}$ | $-2.99_{\pm9.45}$ | $-2.72_{\pm9.56}$ | $-2.88_{\pm9.49}$ | $-2.84_{\pm9.51}$ | $-1.19e3_{\pm2.57e3}$ | $-2.27_{\pm5.93}$ | $-2.17_{\pm6.24}$ | $-2.24_{\pm6.05}$ | $-2.26_{\pm6.00}$ | $-1.31e3_{\pm2.08e3}$ | $-5.84_{\pm11.58}$ | $-6.58_{\pm14.02}$ | $-6.18_{\pm12.62}$ | $-5.84_{\pm11.60}$ |
| Trimentoring | $-150.42_{\pm11.58}$ | $0.60_{\pm0.05}$ | $0.66_{\pm0.05}$ | $0.63_{\pm0.05}$ | $0.64_{\pm0.05}$ | $-149.52_{\pm10.74}$ | $0.59_{\pm0.05}$ | $0.64_{\pm0.04}$ | $0.61_{\pm0.04}$ | $0.60_{\pm0.04}$ | $-150.44_{\pm11.39}$ | $0.58_{\pm0.04}$ | $0.64_{\pm0.04}$ | $0.61_{\pm0.04}$ | $0.58_{\pm0.04}$ |

Table 18: Overall results for GTOPX unconstrained tasks with 128 solutions and 50th percentile evaluations. In this case, 0% of the values are missing near the worst value and another 20% near the optimal value. Details are the same as Table 5.

| Steps | | | t = 50 | | | | | t = 100 | | | | | t = 150 | | |
|---|---|---|---|---|---|---|---|---|---|---|---|---|---|---|---|
| Task $f(x^*_{OFF})$ | | | | | | | | GTOPX 2 $-175.10_{\pm 0.75}$ | | | | | | | |
| Metric | FS | SI | OI | SO | SO$_\omega$ | FS | SI | OI | SO | SO$_\omega$ | FS | SI | OI | SO | SO$_\omega$ |
| ARCOO | $-175.69_{\pm 0.87}$ | $0.99_{\pm 0.00}$ | $0.99_{\pm 0.00}$ | $0.99_{\pm 0.00}$ | $0.99_{\pm 0.00}$ | $-175.68_{\pm 0.87}$ | $0.99_{\pm 0.01}$ | $0.99_{\pm 0.00}$ | $0.99_{\pm 0.00}$ | $0.99_{\pm 0.00}$ | $-175.71_{\pm 0.84}$ | $0.99_{\pm 0.01}$ | $0.99_{\pm 0.00}$ | $0.99_{\pm 0.00}$ | $0.99_{\pm 0.01}$ |
| BO | $-374.89_{\pm 74.19}$ | $-0.62_{\pm 0.15}$ | $-0.62_{\pm 0.15}$ | $-0.62_{\pm 0.15}$ | $-0.62_{\pm 0.15}$ | $-372.25_{\pm 22.05}$ | $-0.66_{\pm 0.11}$ | $-0.66_{\pm 0.11}$ | $-0.66_{\pm 0.11}$ | $-0.66_{\pm 0.11}$ | $-353.82_{\pm 56.12}$ | $-0.67_{\pm 0.09}$ | $-0.67_{\pm 0.09}$ | $-0.67_{\pm 0.09}$ | $-0.67_{\pm 0.09}$ |
| CBAS | $-175.10_{\pm 0.75}$ | $1.00_{\pm 0.00}$ | $1.00_{\pm 0.00}$ | $1.00_{\pm 0.00}$ | $1.00_{\pm 0.00}$ | $-175.10_{\pm 0.75}$ | $1.00_{\pm 0.00}$ | $1.00_{\pm 0.00}$ | $1.00_{\pm 0.00}$ | $1.00_{\pm 0.00}$ | $-175.10_{\pm 0.75}$ | $1.00_{\pm 0.00}$ | $1.00_{\pm 0.00}$ | $1.00_{\pm 0.00}$ | $1.00_{\pm 0.00}$ |
| CCDDEA | $-317.67_{\pm 261.19}$ | $0.33_{\pm 0.27}$ | $0.55_{\pm 0.47}$ | $0.41_{\pm 0.34}$ | $0.45_{\pm 0.37}$ | $-1.99e3_{\pm 5.02e3}$ | $-1.99_{\pm 3.72}$ | $-3.55_{\pm 6.69}$ | $-2.54_{\pm 4.76}$ | $-2.33_{\pm 4.36}$ | $-2.04e3_{\pm 3.00e3}$ | $-4.38_{\pm 7.77}$ | $-7.69_{\pm 13.53}$ | $-5.57_{\pm 9.83}$ | $-4.39_{\pm 7.79}$ |
| CMAES | $-142.89_{\pm 3.52}$ | $0.95_{\pm 0.01}$ | $1.25_{\pm 0.04}$ | $1.08_{\pm 0.01}$ | $1.13_{\pm 0.02}$ | - | - | - | - | - | - | - | - | - | - |
| TTDDEA | $-1.51e3_{\pm 3.28e3}$ | $-8.64_{\pm 22.64}$ | $-8.64_{\pm 22.64}$ | $-8.64_{\pm 22.64}$ | $-8.64_{\pm 22.64}$ | $-3.23e3_{\pm 7.88e3}$ | $-13.88_{\pm 36.83}$ | $-13.87_{\pm 36.84}$ | $-13.88_{\pm 36.83}$ | $-13.88_{\pm 36.83}$ | $-3.90e3_{\pm 9.52e3}$ | $-17.74_{\pm 46.99}$ | $-17.73_{\pm 46.99}$ | $-17.73_{\pm 46.99}$ | $-17.74_{\pm 46.99}$ |
| Trimentoring | $-144.77_{\pm 6.36}$ | $0.93_{\pm 0.03}$ | $1.25_{\pm 0.05}$ | $1.06_{\pm 0.03}$ | $1.12_{\pm 0.04}$ | $-147.56_{\pm 4.51}$ | $0.92_{\pm 0.03}$ | $1.24_{\pm 0.05}$ | $1.06_{\pm 0.03}$ | $1.01_{\pm 0.03}$ | $-149.87_{\pm 4.13}$ | $0.91_{\pm 0.03}$ | $1.23_{\pm 0.04}$ | $1.05_{\pm 0.03}$ | $0.92_{\pm 0.03}$ |
| Task $f(x^*_{OFF})$ | | | | | | | | GTOPX 3 $-134.83_{\pm 0.56}$ | | | | | | | |
| ARCOO | $-135.91_{\pm 0.68}$ | $0.99_{\pm 0.00}$ | $0.99_{\pm 0.00}$ | $0.99_{\pm 0.00}$ | $0.99_{\pm 0.00}$ | $-135.99_{\pm 0.65}$ | $0.99_{\pm 0.00}$ | $0.99_{\pm 0.00}$ | $0.99_{\pm 0.00}$ | $0.99_{\pm 0.00}$ | $-135.93_{\pm 0.60}$ | $0.99_{\pm 0.00}$ | $0.99_{\pm 0.00}$ | $0.99_{\pm 0.00}$ | $0.99_{\pm 0.00}$ |
| BO | $-361.59_{\pm 95.92}$ | $-1.36_{\pm 0.28}$ | $-1.36_{\pm 0.28}$ | $-1.36_{\pm 0.28}$ | $-1.36_{\pm 0.28}$ | $-283.12_{\pm 43.75}$ | $-1.34_{\pm 0.21}$ | $-1.34_{\pm 0.21}$ | $-1.34_{\pm 0.21}$ | $-1.34_{\pm 0.21}$ | $-376.63_{\pm 34.76}$ | $-1.35_{\pm 0.17}$ | $-1.35_{\pm 0.17}$ | $-1.35_{\pm 0.17}$ | $-1.35_{\pm 0.17}$ |
| CBAS | $-134.83_{\pm 0.56}$ | $1.00_{\pm 0.00}$ | $1.00_{\pm 0.00}$ | $1.00_{\pm 0.00}$ | $1.00_{\pm 0.00}$ | $-134.83_{\pm 0.56}$ | $1.00_{\pm 0.00}$ | $1.00_{\pm 0.00}$ | $1.00_{\pm 0.00}$ | $1.00_{\pm 0.00}$ | $-134.83_{\pm 0.56}$ | $1.00_{\pm 0.00}$ | $1.00_{\pm 0.00}$ | $1.00_{\pm 0.00}$ | $1.00_{\pm 0.00}$ |
| CCDDEA | $-4.73e5_{\pm 1.24e6}$ | $-1.13e3_{\pm 2.71e3}$ | $-1.19e3_{\pm 2.79e3}$ | $-1.16e3_{\pm 2.76e3}$ | $-1.17e3_{\pm 2.76e3}$ | $-623.27_{\pm 1.12e3}$ | $-547.41_{\pm 1.26e3}$ | $-672.37_{\pm 1.50e3}$ | $-601.67_{\pm 1.38e3}$ | $-582.66_{\pm 1.35e3}$ | $-627.90_{\pm 1.52e3}$ | $-364.42_{\pm 848.71}$ | $-448.25_{\pm 996.81}$ | $-400.75_{\pm 917.11}$ | $-364.85_{\pm 849.55}$ |
| CMAES | $-1.72e3_{\pm 2.97e5}$ | $-0.82_{\pm 2.94}$ | $-0.88_{\pm 3.20}$ | $-0.85_{\pm 3.07}$ | $-0.86_{\pm 5.11}$ | - | - | - | - | - | - | - | - | - | - |
| TTDDEA | $-4.99e7_{\pm 1.08e8}$ | $-2.12e5_{\pm 5.16e5}$ | $-2.12e5_{\pm 5.16e5}$ | $-2.12e5_{\pm 5.16e5}$ | $-2.12e5_{\pm 5.16e5}$ | $-7.76e6_{\pm 1.30e7}$ | $-3.52e5_{\pm 8.20e5}$ | $-3.52e5_{\pm 8.20e5}$ | $-3.52e5_{\pm 8.20e5}$ | $-3.52e5_{\pm 8.20e5}$ | $-1.20e7_{\pm 2.01e7}$ | $-2.65e6_{\pm 6.28e6}$ | $-2.65e6_{\pm 6.28e6}$ | $-2.65e6_{\pm 6.28e6}$ | $-2.65e6_{\pm 6.28e6}$ |
| Trimentoring | $-135.22_{\pm 3.64}$ | $-11.61_{\pm 33.10}$ | $-11.51_{\pm 33.10}$ | $-11.56_{\pm 33.08}$ | $-11.55_{\pm 33.08}$ | $-134.71_{\pm 3.54}$ | $-11.67_{\pm 33.23}$ | $-11.67_{\pm 33.23}$ | $-11.63_{\pm 33.25}$ | $-11.64_{\pm 33.24}$ | $-133.42_{\pm 3.45}$ | $-11.69_{\pm 33.29}$ | $-11.60_{\pm 33.33}$ | $-11.65_{\pm 33.31}$ | $-11.69_{\pm 33.29}$ |
| Task $f(x^*_{OFF})$ | | | | | | | | GTOPX 4 $-102.96_{\pm 0.29}$ | | | | | | | |
| ARCOO | $-419.23_{\pm 276.47}$ | $-0.02_{\pm 0.77}$ | $-0.02_{\pm 0.77}$ | $-0.02_{\pm 0.77}$ | $-0.02_{\pm 0.77}$ | $-506.51_{\pm 428.89}$ | $-0.62_{\pm 1.77}$ | $-0.62_{\pm 1.77}$ | $-0.62_{\pm 1.77}$ | $-0.62_{\pm 1.77}$ | $-527.40_{\pm 430.76}$ | $-0.96_{\pm 2.25}$ | $-0.96_{\pm 2.25}$ | $-0.96_{\pm 2.25}$ | $-0.96_{\pm 2.25}$ |
| BO | $-479.81_{\pm 139.13}$ | $-1.84_{\pm 0.28}$ | $-1.84_{\pm 0.28}$ | $-1.84_{\pm 0.28}$ | $-1.84_{\pm 0.28}$ | $-587.99_{\pm 133.55}$ | $-1.77_{\pm 0.21}$ | $-1.77_{\pm 0.21}$ | $-1.77_{\pm 0.21}$ | $-1.77_{\pm 0.21}$ | $-565.72_{\pm 142.23}$ | $-1.74_{\pm 0.15}$ | $-1.74_{\pm 0.15}$ | $-1.74_{\pm 0.15}$ | $-1.74_{\pm 0.15}$ |
| CBAS | $-194.03_{\pm 0.88}$ | $1.00_{\pm 0.00}$ | $1.00_{\pm 0.00}$ | $1.00_{\pm 0.00}$ | $1.00_{\pm 0.00}$ | $-194.03_{\pm 0.88}$ | $1.00_{\pm 0.00}$ | $1.00_{\pm 0.00}$ | $1.00_{\pm 0.00}$ | $1.00_{\pm 0.00}$ | $-194.03_{\pm 0.88}$ | $1.00_{\pm 0.00}$ | $1.00_{\pm 0.00}$ | $1.00_{\pm 0.00}$ | $1.00_{\pm 0.00}$ |
| CCDDEA | $-1.47e6_{\pm 1.86e6}$ | $-753.87_{\pm 979.88}$ | $-987.23_{\pm 1.15e3}$ | $-852.62_{\pm 1.14e3}$ | $-893.18_{\pm 1.20e3}$ | $-3.69e6_{\pm 6.12e6}$ | $-4.23e3_{\pm 6.05e3}$ | $-5.35e3_{\pm 7.44e3}$ | $-4.72e3_{\pm 6.67e3}$ | $-4.55e3_{\pm 6.46e3}$ | $-2.99e6_{\pm 4.26e6}$ | $-1.03e4_{\pm 1.38e4}$ | $-1.27e4_{\pm 1.71e4}$ | $-1.14e4_{\pm 1.53e4}$ | $-1.03e4_{\pm 1.38e4}$ |
| CMAES | $-8.11e3_{\pm 2.01e4}$ | $-2.51_{\pm 6.72}$ | $-2.51_{\pm 6.72}$ | $-2.51_{\pm 6.72}$ | $-2.51_{\pm 6.72}$ | - | - | - | - | - | - | - | - | - | - |
| TTDDEA | $-2.76e4_{\pm 3.96e4}$ | $-415.05_{\pm 1.07e3}$ | $-416.28_{\pm 1.07e3}$ | $-415.63_{\pm 1.07e3}$ | $-415.84_{\pm 1.07e3}$ | $-8.57e4_{\pm 2.00e5}$ | $-842.77_{\pm 2.15e3}$ | $-845.93_{\pm 2.15e3}$ | $-844.25_{\pm 2.15e3}$ | $-843.75_{\pm 2.15e3}$ | $-2.64e4_{\pm 6.65e4}$ | $-676.32_{\pm 1.69e3}$ | $-680.34_{\pm 1.69e3}$ | $-678.20_{\pm 1.69e3}$ | $-676.34_{\pm 1.69e3}$ |
| Trimentoring | $-422.30_{\pm 160.70}$ | $-0.75_{\pm 1.43}$ | $-0.76_{\pm 1.43}$ | $-0.75_{\pm 1.43}$ | $-0.76_{\pm 1.43}$ | $-459.20_{\pm 242.22}$ | $-0.87_{\pm 1.58}$ | $-0.88_{\pm 1.57}$ | $-0.88_{\pm 1.57}$ | $-0.87_{\pm 1.58}$ | $-501.03_{\pm 335.76}$ | $-1.05_{\pm 1.95}$ | $-1.07_{\pm 1.94}$ | $-1.06_{\pm 1.95}$ | $-1.05_{\pm 1.95}$ |
| Task $f(x^*_{OFF})$ | | | | | | | | GTOPX 6 $-194.03_{\pm 0.88}$ | | | | | | | |
| ARCOO | $-106.27_{\pm 3.12}$ | $0.98_{\pm 0.01}$ | $0.98_{\pm 0.01}$ | $0.98_{\pm 0.01}$ | $0.98_{\pm 0.01}$ | $-107.64_{\pm 5.02}$ | $0.96_{\pm 0.03}$ | $0.96_{\pm 0.03}$ | $0.96_{\pm 0.03}$ | $0.96_{\pm 0.03}$ | $-107.99_{\pm 5.64}$ | $0.95_{\pm 0.04}$ | $0.95_{\pm 0.04}$ | $0.95_{\pm 0.04}$ | $0.95_{\pm 0.04}$ |
| BO | $-161.50_{\pm 14.77}$ | $0.01_{\pm 0.06}$ | $0.01_{\pm 0.06}$ | $0.01_{\pm 0.06}$ | $0.01_{\pm 0.06}$ | $-184.04_{\pm 32.00}$ | $0.02_{\pm 0.05}$ | $0.02_{\pm 0.05}$ | $0.02_{\pm 0.05}$ | $0.02_{\pm 0.05}$ | $-167.23_{\pm 32.42}$ | $0.02_{\pm 0.06}$ | $0.02_{\pm 0.06}$ | $0.02_{\pm 0.06}$ | $0.02_{\pm 0.06}$ |
| CBAS | $-102.96_{\pm 0.29}$ | $1.00_{\pm 0.00}$ | $1.00_{\pm 0.00}$ | $1.00_{\pm 0.00}$ | $1.00_{\pm 0.00}$ | $-102.96_{\pm 0.29}$ | $1.00_{\pm 0.00}$ | $1.00_{\pm 0.00}$ | $1.00_{\pm 0.00}$ | $1.00_{\pm 0.00}$ | $-102.96_{\pm 0.29}$ | $1.00_{\pm 0.00}$ | $1.00_{\pm 0.00}$ | $1.00_{\pm 0.00}$ | $1.00_{\pm 0.00}$ |
| CCDDEA | $-129.25_{\pm 60.89}$ | $0.14_{\pm 0.22}$ | $0.27_{\pm 0.37}$ | $0.19_{\pm 0.27}$ | $0.21_{\pm 0.30}$ | $-6.79e3_{\pm 1.75e4}$ | $-15.31_{\pm 40.34}$ | $-30.28_{\pm 79.86}$ | $-20.34_{\pm 53.60}$ | $-18.38_{\pm 48.44}$ | $-6.79e3_{\pm 1.75e4}$ | $-30.59_{\pm 80.60}$ | $-60.50_{\pm 159.53}$ | $-40.63_{\pm 107.10}$ | $-30.68_{\pm 80.87}$ |
| CMAES | $-149.68_{\pm 7.55}$ | $0.33_{\pm 0.10}$ | $0.33_{\pm 0.10}$ | $0.33_{\pm 0.10}$ | $0.33_{\pm 0.10}$ | - | - | - | - | - | - | - | - | - | - |
| TTDDEA | $-149.98_{\pm 85.52}$ | $-0.27_{\pm 2.32}$ | $-0.03_{\pm 2.41}$ | $-0.17_{\pm 2.35}$ | $-0.13_{\pm 2.37}$ | $-134.31_{\pm 41.81}$ | $0.05_{\pm 1.24}$ | $0.27_{\pm 1.35}$ | $0.14_{\pm 1.28}$ | $0.10_{\pm 1.27}$ | $-159.00_{\pm 68.59}$ | $0.13_{\pm 0.80}$ | $0.31_{\pm 0.92}$ | $0.20_{\pm 0.85}$ | $0.13_{\pm 0.80}$ |
| Trimentoring | $-134.17_{\pm 24.85}$ | $-17.88_{\pm 23.65}$ | $-17.88_{\pm 23.65}$ | $-17.88_{\pm 23.65}$ | $-17.88_{\pm 23.65}$ | $-133.79_{\pm 24.07}$ | $-18.00_{\pm 23.75}$ | $-18.00_{\pm 23.76}$ | $-18.00_{\pm 23.76}$ | $-18.00_{\pm 23.76}$ | $-133.88_{\pm 24.09}$ | $-18.04_{\pm 23.79}$ | $-18.04_{\pm 23.79}$ | $-18.04_{\pm 23.79}$ | $-18.04_{\pm 23.79}$ |

Table 19: Overall results for GTOPX unconstrained tasks with 128 solutions and 50th percentile evaluations. In this case, 0% of the values are missing near the worst value and another 10% near the optimal value. Details are the same as Table 5.

| Steps | | | t = 50 | | | | | t = 100 | | | | | t = 150 | | |
|---|---|---|---|---|---|---|---|---|---|---|---|---|---|---|---|
| Task $f(x^*_{OFF})$ | | | | | | | | GTOPX 2 $-136.28_{\pm 0.39}$ | | | | | | | |
| Metric | FS | SI | OI | SO | SO$_\omega$ | FS | SI | OI | SO | SO$_\omega$ | FS | SI | OI | SO | SO$_\omega$ |
| ARCOO | $-137.65_{\pm 0.47}$ | $0.98_{\pm 0.00}$ | $0.98_{\pm 0.00}$ | $0.98_{\pm 0.00}$ | $0.98_{\pm 0.00}$ | $-137.61_{\pm 0.47}$ | $0.98_{\pm 0.00}$ | $0.98_{\pm 0.00}$ | $0.98_{\pm 0.00}$ | $0.98_{\pm 0.00}$ | $-137.64_{\pm 0.44}$ | $0.98_{\pm 0.00}$ | $0.98_{\pm 0.00}$ | $0.98_{\pm 0.00}$ | $0.98_{\pm 0.00}$ |
| BO | $-345.41_{\pm 40.43}$ | $-1.73_{\pm 0.14}$ | $-1.73_{\pm 0.14}$ | $-1.73_{\pm 0.14}$ | $-1.73_{\pm 0.14}$ | $-425.87_{\pm 95.13}$ | $-1.72_{\pm 0.12}$ | $-1.72_{\pm 0.12}$ | $-1.72_{\pm 0.12}$ | $-1.72_{\pm 0.12}$ | $-371.23_{\pm 130.08}$ | $-1.73_{\pm 0.11}$ | $-1.73_{\pm 0.11}$ | $-1.73_{\pm 0.11}$ | $-1.73_{\pm 0.11}$ |
| CBAS | $-136.28_{\pm 0.39}$ | $1.00_{\pm 0.00}$ | $1.00_{\pm 0.00}$ | $1.00_{\pm 0.00}$ | $1.00_{\pm 0.00}$ | $-136.28_{\pm 0.39}$ | $1.00_{\pm 0.00}$ | $1.00_{\pm 0.00}$ | $1.00_{\pm 0.00}$ | $1.00_{\pm 0.00}$ | $-136.28_{\pm 0.39}$ | $1.00_{\pm 0.00}$ | $1.00_{\pm 0.00}$ | $1.00_{\pm 0.00}$ | $1.00_{\pm 0.00}$ |
| CCDDEA | $-289.62_{\pm 199.21}$ | $-0.65_{\pm 2.31}$ | $-0.45_{\pm 2.42}$ | $-0.58_{\pm 2.34}$ | $-0.55_{\pm 2.36}$ | $-557.27_{\pm 519.51}$ | $-0.99_{\pm 1.28}$ | $-1.47_{\pm 1.69}$ | $-1.15_{\pm 1.39}$ | $-1.09_{\pm 1.34}$ | $-553.40_{\pm 522.57}$ | $-1.42_{\pm 1.66}$ | $-2.44_{\pm 3.14}$ | $-1.77_{\pm 2.14}$ | $-1.42_{\pm 1.66}$ |
| CMAES | $-139.68_{\pm 4.48}$ | $0.90_{\pm 0.03}$ | $0.95_{\pm 0.07}$ | $0.93_{\pm 0.05}$ | $0.93_{\pm 0.05}$ | - | - | - | - | - | - | - | - | - | - |
| TTDDEA | $-342.66_{\pm 176.59}$ | $-2.12_{\pm 1.28}$ | $-2.35_{\pm 1.58}$ | $-2.22_{\pm 2.40}$ | $-2.26_{\pm 2.45}$ | $-481.85_{\pm 430.98}$ | $-2.50_{\pm 2.92}$ | $-2.85_{\pm 3.56}$ | $-2.65_{\pm 3.17}$ | $-2.60_{\pm 3.08}$ | $-447.38_{\pm 349.03}$ | $-2.90_{\pm 3.56}$ | $-3.36_{\pm 4.47}$ | $-3.09_{\pm 3.92}$ | $-2.90_{\pm 3.56}$ |
| Trimentoring | $-138.96_{\pm 3.41}$ | $0.91_{\pm 0.07}$ | $0.98_{\pm 0.04}$ | $0.94_{\pm 0.06}$ | $0.95_{\pm 0.05}$ | $-139.33_{\pm 4.74}$ | $0.90_{\pm 0.08}$ | $0.97_{\pm 0.08}$ | $0.93_{\pm 0.06}$ | $0.92_{\pm 0.07}$ | $-139.93_{\pm 3.85}$ | $0.90_{\pm 0.06}$ | $0.96_{\pm 0.05}$ | $0.93_{\pm 0.07}$ | $0.90_{\pm 0.06}$ |
| Task $f(x^*_{OFF})$ | | | | | | | | GTOPX 3 $-102.69_{\pm 0.49}$ | | | | | | | |
| ARCOO | $-104.67_{\pm 0.77}$ | $0.97_{\pm 0.01}$ | $0.97_{\pm 0.01}$ | $0.97_{\pm 0.01}$ | $0.97_{\pm 0.01}$ | $-104.70_{\pm 0.93}$ | $0.97_{\pm 0.01}$ | $0.97_{\pm 0.01}$ | $0.97_{\pm 0.01}$ | $0.97_{\pm 0.01}$ | $-104.97_{\pm 1.05}$ | $0.97_{\pm 0.01}$ | $0.97_{\pm 0.01}$ | $0.97_{\pm 0.01}$ | $0.97_{\pm 0.01}$ |
| BO | $-307.64_{\pm 40.77}$ | $-2.66_{\pm 0.44}$ | $-2.66_{\pm 0.44}$ | $-2.66_{\pm 0.44}$ | $-2.66_{\pm 0.44}$ | $-339.86_{\pm 122.60}$ | $-2.61_{\pm 0.28}$ | $-2.61_{\pm 0.28}$ | $-2.61_{\pm 0.28}$ | $-2.61_{\pm 0.28}$ | $-312.77_{\pm 109.25}$ | $-2.73_{\pm 0.28}$ | $-2.73_{\pm 0.28}$ | $-2.73_{\pm 0.28}$ | $-2.73_{\pm 0.28}$ |
| CBAS | $-102.69_{\pm 0.49}$ | $1.00_{\pm 0.00}$ | $1.00_{\pm 0.00}$ | $1.00_{\pm 0.00}$ | $1.00_{\pm 0.00}$ | $-102.69_{\pm 0.49}$ | $1.00_{\pm 0.00}$ | $1.00_{\pm 0.00}$ | $1.00_{\pm 0.00}$ | $1.00_{\pm 0.00}$ | $-102.69_{\pm 0.49}$ | $1.00_{\pm 0.00}$ | $1.00_{\pm 0.00}$ | $1.00_{\pm 0.00}$ | $1.00_{\pm 0.00}$ |
| CCDDEA | $-8.22e4_{\pm 2.13e5}$ | $-247.15_{\pm 550.17}$ | $-260.52_{\pm 572.54}$ | $-253.40_{\pm 561.21}$ | $-255.68_{\pm 565.01}$ | $-153.35_{\pm 57.09}$ | $-170.41_{\pm 182.89}$ | $-155.07_{\pm 302.14}$ | $-122.59_{\pm 227.47}$ | $-115.22_{\pm 210.70}$ | $-170.41_{\pm 71.58}$ | $-68.45_{\pm 121.51}$ | $-103.11_{\pm 200.72}$ | $-81.52_{\pm 155.18}$ | $-68.59_{\pm 121.82}$ |
| CMAES | $-1.44e3_{\pm 3.41e3}$ | $-4.84_{\pm 14.46}$ | $-4.84_{\pm 14.46}$ | $-4.84_{\pm 14.46}$ | $-4.84_{\pm 14.46}$ | - | - | - | - | - | - | - | - | - | - |
| TTDDEA | $-2.48e6_{\pm 5.43e6}$ | $-3.97e4_{\pm 5.14e4}$ | $-3.97e4_{\pm 5.14e4}$ | $-3.97e4_{\pm 5.14e4}$ | $-3.97e4_{\pm 5.14e4}$ | $-4.28e7_{\pm 6.84e7}$ | $-4.57e5_{\pm 4.39e5}$ | $-4.57e5_{\pm 4.39e5}$ | $-4.57e5_{\pm 4.39e5}$ | $-4.57e5_{\pm 4.39e5}$ | $-1.91e8_{\pm 4.76e8}$ | $-1.28e6_{\pm 1.68e6}$ | $-1.28e6_{\pm 1.68e6}$ | $-1.28e6_{\pm 1.68e6}$ | $-1.28e6_{\pm 1.68e6}$ |
| Trimentoring | $-136.98_{\pm 19.06}$ | $0.56_{\pm 0.20}$ | $0.57_{\pm 0.20}$ | $0.57_{\pm 0.20}$ | $0.57_{\pm 0.20}$ | $-145.03_{\pm 34.18}$ | $0.49_{\pm 0.29}$ | $0.50_{\pm 0.29}$ | $0.50_{\pm 0.29}$ | $0.50_{\pm 0.29}$ | $-154.51_{\pm 59.20}$ | $0.44_{\pm 0.41}$ | $0.44_{\pm 0.41}$ | $0.44_{\pm 0.41}$ | $0.44_{\pm 0.41}$ |
| Task $f(x^*_{OFF})$ | | | | | | | | GTOPX 4 $-153.62_{\pm 0.60}$ | | | | | | | |
| ARCOO | $-335.23_{\pm 34.53}$ | $-1.24_{\pm 0.52}$ | $-1.24_{\pm 0.52}$ | $-1.24_{\pm 0.52}$ | $-1.24_{\pm 0.52}$ | $-337.83_{\pm 34.94}$ | $-1.32_{\pm 0.56}$ | $-1.32_{\pm 0.56}$ | $-1.32_{\pm 0.56}$ | $-1.32_{\pm 0.56}$ | $-339.75_{\pm 31.77}$ | $-1.35_{\pm 0.56}$ | $-1.35_{\pm 0.56}$ | $-1.35_{\pm 0.56}$ | $-1.35_{\pm 0.56}$ |
| BO | $-533.72_{\pm 159.31}$ | $-3.51_{\pm 0.65}$ | $-3.51_{\pm 0.65}$ | $-3.51_{\pm 0.65}$ | $-3.51_{\pm 0.65}$ | $-534.44_{\pm 219.41}$ | $-3.56_{\pm 0.46}$ | $-3.56_{\pm 0.46}$ | $-3.56_{\pm 0.46}$ | $-3.56_{\pm 0.46}$ | $-471.34_{\pm 147.83}$ | $-3.58_{\pm 0.41}$ | $-3.58_{\pm 0.41}$ | $-3.58_{\pm 0.41}$ | $-3.58_{\pm 0.41}$ |
| CBAS | $-153.62_{\pm 0.60}$ | $1.00_{\pm 0.00}$ | $1.00_{\pm 0.00}$ | $1.00_{\pm 0.00}$ | $1.00_{\pm 0.00}$ | $-153.62_{\pm 0.60}$ | $1.00_{\pm 0.00}$ | $1.00_{\pm 0.00}$ | $1.00_{\pm 0.00}$ | $1.00_{\pm 0.00}$ | $-153.62_{\pm 0.60}$ | $1.00_{\pm 0.00}$ | $1.00_{\pm 0.00}$ | $1.00_{\pm 0.00}$ | $1.00_{\pm 0.00}$ |
| CCDDEA | $-2.60e3_{\pm 3.94e3}$ | $-466.30_{\pm 592.36}$ | $-478.11_{\pm 583.64}$ | $-471.64_{\pm 583.09}$ | $-473.66_{\pm 583.31}$ | $-1.04e8_{\pm 2.52e8}$ | $-3.75e5_{\pm 8.47e5}$ | $-3.86e5_{\pm 8.44e5}$ | $-3.79e5_{\pm 8.45e5}$ | $-3.78e5_{\pm 8.46e5}$ | $-1.65e7_{\pm 2.33e7}$ | $-3.92e5_{\pm 7.51e5}$ | $-4.13e5_{\pm 7.46e5}$ | $-4.01e5_{\pm 7.50e5}$ | $-3.92e5_{\pm 7.51e5}$ |
| CMAES | $-333.46_{\pm 43.49}$ | $-0.52_{\pm 0.22}$ | $-0.52_{\pm 0.22}$ | $-0.52_{\pm 0.22}$ | $-0.52_{\pm 0.22}$ | - | - | - | - | - | - | - | - | - | - |
| TTDDEA | $-7.90e4_{\pm 2.08e5}$ | $-369.40_{\pm 960.09}$ | $-369.91_{\pm 960.01}$ | $-369.70_{\pm 960.98}$ | $-369.70_{\pm 960.98}$ | $-1.11e5_{\pm 2.93e5}$ | $-2.80e3_{\pm 7.40e3}$ | $-2.80e3_{\pm 7.40e3}$ | $-2.80e3_{\pm 7.40e3}$ | $-2.80e3_{\pm 7.40e3}$ | $-1.14e8_{\pm 3.01e8}$ | $-2.30e5_{\pm 6.10e5}$ | $-2.30e5_{\pm 6.10e5}$ | $-2.30e5_{\pm 6.10e5}$ | $-2.30e5_{\pm 6.10e5}$ |
| Trimentoring | $-356.75_{\pm 19.23}$ | $-1.19_{\pm 0.28}$ | $-1.19_{\pm 0.28}$ | $-1.19_{\pm 0.28}$ | $-1.19_{\pm 0.28}$ | $-362.71_{\pm 14.37}$ | $-1.30_{\pm 0.28}$ | $-1.30_{\pm 0.26}$ | $-1.30_{\pm 0.28}$ | $-1.30_{\pm 0.28}$ | $-372.42_{\pm 12.89}$ | $-1.37_{\pm 0.26}$ | $-1.37_{\pm 0.26}$ | $-1.37_{\pm 0.26}$ | $-1.37_{\pm 0.26}$ |
| Task $f(x^*_{OFF})$ | | | | | | | | GTOPX 6 $-83.37_{\pm 0.23}$ | | | | | | | |
| ARCOO | $-97.07_{\pm 10.48}$ | $0.88_{\pm 0.09}$ | $0.88_{\pm 0.09}$ | $0.88_{\pm 0.09}$ | $0.88_{\pm 0.09}$ | $-97.90_{\pm 10.46}$ | $0.79_{\pm 0.15}$ | $0.79_{\pm 0.15}$ | $0.79_{\pm 0.15}$ | $0.79_{\pm 0.15}$ | $-99.07_{\pm 11.70}$ | $0.76_{\pm 0.18}$ | $0.76_{\pm 0.18}$ | $0.76_{\pm 0.18}$ | $0.76_{\pm 0.18}$ |
| BO | $-162.46_{\pm 25.58}$ | $-0.78_{\pm 0.18}$ | $-0.78_{\pm 0.18}$ | $-0.78_{\pm 0.18}$ | $-0.78_{\pm 0.18}$ | $-186.54_{\pm 35.96}$ | $-0.82_{\pm 0.14}$ | $-0.82_{\pm 0.14}$ | $-0.82_{\pm 0.14}$ | $-0.82_{\pm 0.14}$ | $-174.09_{\pm 16.79}$ | $-0.82_{\pm 0.13}$ | $-0.82_{\pm 0.13}$ | $-0.82_{\pm 0.13}$ | $-0.82_{\pm 0.13}$ |
| CBAS | $-83.37_{\pm 0.23}$ | $1.00_{\pm 0.00}$ | $1.00_{\pm 0.00}$ | $1.00_{\pm 0.00}$ | $1.00_{\pm 0.00}$ | $-83.37_{\pm 0.23}$ | $1.00_{\pm 0.00}$ | $1.00_{\pm 0.00}$ | $1.00_{\pm 0.00}$ | $1.00_{\pm 0.00}$ | $-83.37_{\pm 0.23}$ | $1.00_{\pm 0.00}$ | $1.00_{\pm 0.00}$ | $1.00_{\pm 0.00}$ | $1.00_{\pm 0.00}$ |
| CCDDEA | $-223.46_{\pm 177.06}$ | $-0.79_{\pm 0.67}$ | $-1.14_{\pm 1.14}$ | $-0.92_{\pm 0.83}$ | $-0.98_{\pm 0.92}$ | $-151.49_{\pm 65.38}$ | $-0.72_{\pm 0.48}$ | $-1.14_{\pm 0.84}$ | $-0.88_{\pm 0.61}$ | $-0.82_{\pm 0.56}$ | $-155.23_{\pm 60.73}$ | $-0.65_{\pm 0.45}$ | $-0.99_{\pm 0.77}$ | $-0.78_{\pm 0.56}$ | $-0.65_{\pm 0.45}$ |
| CMAES | $-148.32_{\pm 8.01}$ | $-0.30_{\pm 0.18}$ | $-0.30_{\pm 0.18}$ | $-0.30_{\pm 0.18}$ | $-0.30_{\pm 0.18}$ | - | - | - | - | - | - | - | - | - | - |
| TTDDEA | $-133.30_{\pm 42.69}$ | $-0.08_{\pm 0.85}$ | $0.07_{\pm 0.95}$ | $-0.02_{\pm 0.89}$ | $0.00_{\pm 0.91}$ | $-166.56_{\pm 88.16}$ | $-0.29_{\pm 1.08}$ | $-0.19_{\pm 1.16}$ | $-0.25_{\pm 1.11}$ | $-0.27_{\pm 1.10}$ | $-411.53_{\pm 359.65}$ | $-0.86_{\pm 1.60}$ | $-0.78_{\pm 1.67}$ | $-0.83_{\pm 1.63}$ | $-0.86_{\pm 1.60}$ |
| Trimentoring | $-137.76_{\pm 31.30}$ | $-8.70_{\pm 22.85}$ | $-8.70_{\pm 22.85}$ | $-8.70_{\pm 22.85}$ | $-8.70_{\pm 22.85}$ | $-138.06_{\pm 31.55}$ | $-8.79_{\pm 22.95}$ | $-8.79_{\pm 22.95}$ | $-8.79_{\pm 22.95}$ | $-8.79_{\pm 22.95}$ | $-137.65_{\pm 31.44}$ | $-8.82_{\pm 22.99}$ | $-8.82_{\pm 22.99}$ | $-8.82_{\pm 22.99}$ | $-8.82_{\pm 22.99}$ |

Table 20: Overall results for GTOPX unconstrained tasks with 128 solutions and 50th percentile evaluations. In this case, 10% of the values are missing near the worst value and another 40% near the optimal value. Details are the same as Table 5.

| Steps | | t = 50 | | | | | t = 100 | | | | | t = 150 | | | |
|---|---|---|---|---|---|---|---|---|---|---|---|---|---|---|---|
| **Task** $f(x^*_{OFF})$ | | | | | | | | GTOPX 2 −275.27±2.34 | | | | | | | |
| Metric | FS | SI | OI | SO | SO$_\omega$ | FS | SI | OI | SO | SO$_\omega$ | FS | SI | OI | SO | SO$_\omega$ |
| ARCOO | -267.52±21.10 | 0.98±0.03 | 1.01±0.05 | 1.00±0.01 | 1.00±0.02 | -265.67±27.80 | 0.97±0.04 | 1.03±0.07 | 0.99±0.02 | 0.99±0.02 | -264.76±29.86 | 0.97±0.04 | 1.03±0.09 | 1.00±0.01 | 0.97±0.04 |
| BO | -348.94±71.62 | 0.44±0.07 | 0.55±0.09 | 0.49±0.08 | 0.51±0.09 | -346.21±78.31 | 0.45±0.04 | 0.57±0.05 | 0.50±0.04 | 0.44±0.04 | -406.55±135.88 | 0.44±0.04 | 0.57±0.05 | 0.50±0.05 | 0.44±0.04 |
| CBAS | -275.27±2.34 | 1.00±0.00 | 1.00±0.00 | 1.00±0.00 | 1.00±0.00 | -275.27±2.34 | 1.00±0.00 | 1.00±0.00 | 1.00±0.00 | 1.00±0.00 | -275.27±2.34 | 1.00±0.00 | 1.00±0.00 | 1.00±0.00 | 1.00±0.00 |
| CCDDEA | -302.87±103.39 | 0.64±0.12 | 1.12±0.25 | 0.81±0.17 | 0.90±0.19 | -876.36±601.32 | 0.20±0.49 | 0.37±0.85 | 0.26±0.62 | 0.24±0.57 | -724.58±320.70 | -0.12±0.84 | -0.20±1.44 | -0.15±1.06 | -0.12±0.84 |
| CMAES | -148.50±6.33 | 0.97±0.01 | 1.56±0.02 | 1.20±0.01 | 1.30±0.01 | - | - | - | - | - | - | - | - | - | - |
| TTDDEA | -254.56±95.58 | 0.62±0.50 | 1.02±0.67 | 0.78±0.56 | 0.84±0.59 | -217.19±56.05 | 0.66±0.29 | 1.09±0.44 | 0.82±0.34 | 0.76±0.32 | -229.69±66.18 | 0.71±0.19 | 1.14±0.34 | 0.87±0.24 | 0.71±0.19 |
| Trimentoring | -197.51±62.76 | -20.33±56.36 | -19.97±56.50 | -20.19±56.41 | -20.13±56.44 | -196.25±63.87 | -20.44±56.65 | -20.30±56.70 | -20.30±56.70 | -20.36±56.60 | -197.06±63.17 | -20.48±56.75 | -20.11±56.89 | -20.34±56.80 | -20.48±56.75 |
| **Task** $f(x^*_{OFF})$ | | | | | | | | GTOPX 3 −228.84±3.46 | | | | | | | |
| ARCOO | -223.94±8.00 | 0.95±0.05 | 1.03±0.04 | 0.99±0.01 | 1.00±0.01 | -225.68±4.34 | 0.94±0.06 | 1.03±0.04 | 0.98±0.02 | 0.97±0.03 | -223.21±8.86 | 0.94±0.06 | 1.03±0.04 | 0.98±0.02 | 0.94±0.06 |
| BO | -475.10±211.82 | 0.18±0.12 | 0.22±0.16 | 0.20±0.14 | 0.21±0.14 | -359.85±119.95 | 0.17±0.10 | 0.22±0.13 | 0.19±0.11 | 0.18±0.11 | -358.26±139.79 | 0.17±0.08 | 0.22±0.11 | 0.19±0.10 | 0.17±0.08 |
| CBAS | -228.84±3.46 | 1.00±0.00 | 1.00±0.00 | 1.00±0.00 | 1.00±0.00 | -228.84±3.46 | 1.00±0.00 | 1.00±0.00 | 1.00±0.00 | 1.00±0.00 | -228.84±3.46 | 1.00±0.00 | 1.00±0.00 | 1.00±0.00 | 1.00±0.00 |
| CCDDEA | -142.24±49.73 | -1.45±3.21 | -2.73±6.09 | -1.89±4.21 | -2.11±4.70 | -354.49±478.96 | -0.56±1.63 | -1.06±3.08 | -0.73±2.13 | -0.67±1.94 | -352.68±479.38 | -0.33±1.30 | -0.62±2.44 | -0.43±1.70 | -0.34±1.31 |
| CMAES | -131.58±6.80 | 0.97±0.01 | 1.49±0.03 | 1.17±0.01 | 1.26±0.02 | - | - | - | - | - | - | - | - | - | - |
| TTDDEA | -994.64±1.20e3 | -3.82±10.42 | -3.70±10.49 | -3.78±10.44 | -3.76±10.45 | -3.78e4±4.41e4 | -96.04±230.13 | -99.46±229.16 | -97.43±229.72 | -96.93±229.87 | -4.31e6±1.09e7 | -1.38e3±2.65e3 | -2.03e3±4.24e3 | -1.63e3±3.25e3 | -1.38e3±2.66e3 |
| Trimentoring | -152.89±36.19 | -30.66±83.62 | -30.21±83.79 | -30.48±83.69 | -30.40±83.73 | -1.29e4±3.37e4 | -40.21±84.19 | -44.86±87.44 | -42.04±85.22 | -41.36±84.81 | -3.21e4±8.46e4 | -55.73±99.18 | -68.80±122.43 | -60.87±107.50 | -55.79±99.27 |
| **Task** $f(x^*_{OFF})$ | | | | | | | | GTOPX 4 −322.50±1.08 | | | | | | | |
| ARCOO | -325.21±1.55 | 0.99±0.00 | 0.99±0.00 | 0.99±0.00 | 0.99±0.00 | -325.57±1.45 | 0.99±0.00 | 0.99±0.00 | 0.99±0.00 | 0.99±0.00 | -325.62±1.40 | 0.99±0.00 | 0.99±0.00 | 0.99±0.00 | 0.99±0.00 |
| BO | -691.49±386.20 | -0.00±0.14 | -0.00±0.17 | -0.00±0.15 | -0.00±0.15 | -598.96±267.73 | 0.06±0.07 | 0.08±0.10 | 0.07±0.09 | 0.07±0.08 | -805.71±384.93 | 0.07±0.06 | 0.09±0.09 | 0.08±0.07 | 0.07±0.06 |
| CBAS | -322.50±1.08 | 1.00±0.00 | 1.00±0.00 | 1.00±0.00 | 1.00±0.00 | -322.50±1.08 | 1.00±0.00 | 1.00±0.00 | 1.00±0.00 | 1.00±0.00 | -322.50±1.08 | 1.00±0.00 | 1.00±0.00 | 1.00±0.00 | 1.00±0.00 |
| CCDDEA | -528.46±454.31 | -6.76±9.87 | -12.00±17.83 | -8.63±12.71 | -9.53±14.07 | -2.62e5±3.68e5 | -4.37e3±1.03e4 | -8.46e3±2.05e4 | -5.75e3±1.37e4 | -5.22e3±1.23e4 | -9.28e5±1.75e6 | -3.72e3±7.01e3 | -6.80e3±1.36e4 | -4.78e3±9.27e3 | -3.73e3±7.03e3 |
| CMAES | -310.86±16.19 | 0.87±0.02 | 1.04±0.04 | 0.95±0.03 | 0.98±0.04 | - | - | - | - | - | - | - | - | - | - |
| TTDDEA | -6.98e4±1.29e5 | -413.64±714.12 | -509.27±874.50 | -456.44±786.07 | -472.99±813.82 | -1.03e5±2.05e5 | -6.70e3±1.71e4 | -8.09e3±2.06e4 | -7.33e3±1.87e4 | -7.12e3±1.82e4 | -9.33e4±1.62e5 | -4.54e3±1.14e4 | -5.48e3±1.36e4 | -4.96e3±1.25e4 | -4.54e3±1.15e4 |
| Trimentoring | -372.67±10.04 | 0.61±0.03 | 0.89±0.04 | 0.73±0.03 | 0.78±0.04 | -376.14±4.37 | 0.58±0.03 | 0.84±0.03 | 0.69±0.03 | 0.65±0.03 | -372.25±9.06 | 0.57±0.02 | 0.83±0.03 | 0.68±0.02 | 0.57±0.02 |
| **Task** $f(x^*_{OFF})$ | | | | | | | | GTOPX 6 −142.09±0.39 | | | | | | | |
| ARCOO | -142.67±0.36 | 0.99±0.00 | 0.99±0.00 | 0.99±0.00 | 0.99±0.00 | -142.80±0.40 | 0.99±0.00 | 0.99±0.00 | 0.99±0.00 | 0.99±0.00 | -142.91±0.30 | 0.99±0.00 | 0.99±0.00 | 0.99±0.00 | 0.99±0.00 |
| BO | -164.26±18.07 | 0.63±0.05 | 0.72±0.04 | 0.67±0.05 | 0.69±0.05 | -155.51±25.08 | 0.59±0.02 | 0.71±0.03 | 0.65±0.02 | 0.63±0.02 | -171.21±21.92 | 0.58±0.02 | 0.71±0.02 | 0.64±0.02 | 0.59±0.02 |
| CBAS | -142.09±0.39 | 1.00±0.00 | 1.00±0.00 | 1.00±0.00 | 1.00±0.00 | -142.09±0.39 | 1.00±0.00 | 1.00±0.00 | 1.00±0.00 | 1.00±0.00 | -142.09±0.39 | 1.00±0.00 | 1.00±0.00 | 1.00±0.00 | 1.00±0.00 |
| CCDDEA | -228.05±109.46 | 0.06±0.29 | 0.13±0.51 | 0.08±0.37 | 0.09±0.41 | -756.52±699.85 | -1.07±1.58 | -1.93±2.87 | -1.37±2.04 | -1.26±1.86 | -683.38±757.82 | -1.95±2.91 | -3.54±5.29 | -2.51±3.76 | -1.95±2.92 |
| CMAES | -155.91±5.22 | 0.88±0.03 | 0.89±0.04 | 0.88±0.04 | 0.88±0.04 | - | - | - | - | - | - | - | - | - | - |
| TTDDEA | -120.37±39.16 | 0.77±0.09 | 1.26±0.18 | 0.96±0.12 | 1.04±0.13 | -110.17±28.22 | 0.77±0.15 | 1.25±0.26 | 0.95±0.19 | 0.88±0.17 | -121.15±27.84 | 0.77±0.16 | 1.25±0.26 | 0.95±0.19 | 0.77±0.16 |
| Trimentoring | -151.78±9.88 | 0.84±0.14 | 0.94±0.07 | 0.89±0.11 | 0.90±0.10 | -151.78±9.52 | 0.83±0.15 | 0.93±0.07 | 0.87±0.12 | 0.86±0.13 | -151.07±8.74 | 0.83±0.15 | 0.92±0.08 | 0.87±0.12 | 0.83±0.15 |

Table 21: Overall results for GTOPX unconstrained tasks with 128 solutions and 50th percentile evaluations. In this case, 20% of the values are missing near the worst value and another 30% near the optimal value. Details are the same as Table 5.

| Steps | | t = 50 | | | | | t = 100 | | | | | t = 150 | | | |
|---|---|---|---|---|---|---|---|---|---|---|---|---|---|---|---|
| **Task** $f(x^*_{OFF})$ | | | | | | | | GTOPX 2 −218.12±1.78 | | | | | | | |
| Metric | FS | SI | OI | SO | SO$_\omega$ | FS | SI | OI | SO | SO$_\omega$ | FS | SI | OI | SO | SO$_\omega$ |
| ARCOO | -218.65±2.03 | 1.00±0.00 | 1.00±0.00 | 1.00±0.00 | 1.00±0.00 | -218.92±1.95 | 0.99±0.00 | 1.00±0.00 | 0.99±0.00 | 0.99±0.00 | -219.06±2.05 | 0.99±0.00 | 0.99±0.00 | 0.99±0.00 | 0.99±0.00 |
| BO | -327.32±58.01 | 0.09±0.07 | 0.09±0.07 | 0.09±0.07 | 0.09±0.07 | -343.40±83.65 | 0.09±0.07 | 0.09±0.07 | 0.09±0.07 | 0.09±0.07 | -367.06±73.39 | 0.11±0.05 | 0.11±0.05 | 0.11±0.05 | 0.11±0.05 |
| CBAS | -218.12±1.78 | 1.00±0.00 | 1.00±0.00 | 1.00±0.00 | 1.00±0.00 | -218.12±1.78 | 1.00±0.00 | 1.00±0.00 | 1.00±0.00 | 1.00±0.00 | -218.12±1.78 | 1.00±0.00 | 1.00±0.00 | 1.00±0.00 | 1.00±0.00 |
| CCDDEA | -208.14±51.88 | 0.51±0.22 | 0.91±0.42 | 0.65±0.29 | 0.72±0.35 | -1.22e3±1.26e3 | -5.95±13.89 | -9.00±20.15 | -7.15±16.44 | -6.71±15.52 | -824.18±428.28 | -4.65±9.63 | -7.25±15.96 | -5.65±11.40 | -4.66±9.65 |
| CMAES | -147.10±2.23 | 0.96±0.01 | 1.43±0.02 | 1.15±0.00 | 1.23±0.01 | - | - | - | - | - | - | - | - | - | - |
| TTDDEA | -298.89±107.26 | 0.53±0.39 | 0.67±0.49 | 0.59±0.43 | 0.61±0.45 | -281.02±83.43 | 0.44±0.41 | 0.60±0.52 | 0.51±0.46 | 0.48±0.44 | -306.22±141.56 | 0.42±0.44 | 0.59±0.56 | 0.49±0.49 | 0.42±0.44 |
| Trimentoring | -167.85±51.30 | -34.71±61.95 | -34.36±62.15 | -34.57±62.03 | -34.51±62.07 | -167.24±51.78 | -34.89±62.27 | -34.75±62.35 | -34.75±62.35 | -34.80±62.53 | -166.62±52.04 | -34.95±62.38 | -34.60±62.58 | -34.81±62.46 | -34.95±62.38 |
| **Task** $f(x^*_{OFF})$ | | | | | | | | GTOPX 3 −171.54±0.81 | | | | | | | |
| ARCOO | -172.49±0.97 | 0.99±0.00 | 0.99±0.00 | 0.99±0.00 | 0.99±0.00 | -172.51±0.92 | 0.99±0.00 | 0.99±0.00 | 0.99±0.00 | 0.99±0.00 | -172.55±0.81 | 0.99±0.00 | 0.99±0.00 | 0.99±0.00 | 0.99±0.00 |
| BO | -434.43±140.66 | -0.43±0.11 | -0.43±0.10 | -0.43±0.11 | -0.43±0.11 | -401.70±123.24 | -0.43±0.08 | -0.45±0.08 | -0.44±0.08 | -0.44±0.08 | -364.91±93.16 | -0.43±0.07 | -0.45±0.06 | -0.44±0.06 | -0.43±0.07 |
| CBAS | -171.54±0.81 | 1.00±0.00 | 1.00±0.00 | 1.00±0.00 | 1.00±0.00 | -171.54±0.81 | 1.00±0.00 | 1.00±0.00 | 1.00±0.00 | 1.00±0.00 | -171.54±0.81 | 1.00±0.00 | 1.00±0.00 | 1.00±0.00 | 1.00±0.00 |
| CCDDEA | -268.55±305.59 | -0.21±1.56 | -0.39±2.89 | -0.27±2.03 | -0.30±2.25 | -506.71±542.70 | -0.68±1.86 | -1.36±3.65 | -0.91±2.47 | -0.82±2.25 | -611.53±810.05 | -0.83±2.15 | -1.66±4.23 | -1.11±2.85 | -0.83±2.16 |
| CMAES | -134.42±4.49 | 0.95±0.01 | 1.27±0.02 | 1.09±0.01 | 1.14±0.01 | - | - | - | - | - | - | - | - | - | - |
| TTDDEA | -5.65e3±1.38e4 | -55.80±140.40 | -55.60±140.48 | -55.73±140.43 | -55.69±140.45 | -3.67e4±7.82e4 | -113.26±203.09 | -113.55±202.94 | -113.36±203.04 | -113.32±203.06 | -1.30e6±1.76e6 | -3.60e3±9.01e3 | -3.64e3±8.99e3 | -3.62e3±9.00e3 | -3.60e3±9.01e3 |
| Trimentoring | -148.48±9.99 | 0.90±0.03 | 1.18±0.07 | 1.02±0.04 | 1.07±0.05 | -147.90±10.75 | 0.90±0.04 | 1.18±0.07 | 1.02±0.04 | 0.98±0.04 | -146.43±9.46 | 0.90±0.04 | 1.18±0.07 | 1.02±0.04 | 0.90±0.04 |
| **Task** $f(x^*_{OFF})$ | | | | | | | | GTOPX 4 −241.61±1.39 | | | | | | | |
| ARCOO | -243.01±1.33 | 0.99±0.00 | 0.99±0.00 | 0.99±0.00 | 0.99±0.00 | -243.54±1.42 | 0.99±0.00 | 0.99±0.00 | 0.99±0.00 | 0.99±0.00 | -243.80±1.43 | 0.99±0.00 | 0.99±0.00 | 0.99±0.00 | 0.99±0.00 |
| BO | -656.30±391.87 | -0.71±0.32 | -0.74±0.35 | -0.72±0.33 | -0.73±0.34 | -666.10±441.83 | -0.77±0.23 | -0.81±0.23 | -0.79±0.23 | -0.78±0.23 | -456.40±259.92 | -0.73±0.14 | -0.77±0.14 | -0.75±0.14 | -0.73±0.14 |
| CBAS | -241.61±1.39 | 1.00±0.00 | 1.00±0.00 | 1.00±0.00 | 1.00±0.00 | -241.61±1.39 | 1.00±0.00 | 1.00±0.00 | 1.00±0.00 | 1.00±0.00 | -241.61±1.39 | 1.00±0.00 | 1.00±0.00 | 1.00±0.00 | 1.00±0.00 |
| CCDDEA | -839.42±612.96 | -14.64±16.52 | -20.68±24.79 | -16.93±19.37 | -17.98±20.79 | -1.86e5±2.65e5 | -259.94±360.15 | -363.51±479.09 | -302.38±411.15 | -287.06±393.08 | -1.87e5±2.65e5 | -511.76±732.75 | -714.61±978.66 | -595.16±837.85 | -512.71±733.97 |
| CMAES | -306.56±11.71 | 0.59±0.05 | 0.59±0.05 | 0.59±0.05 | 0.59±0.05 | - | - | - | - | - | - | - | - | - | - |
| TTDDEA | -1.78e3±3.89e3 | -0.52±2.50 | -0.57±3.07 | -0.55±2.75 | -0.56±2.85 | -3.56e3±8.56e3 | -6.55±18.41 | -7.90±22.37 | -7.17±20.19 | -6.96±19.58 | -4.02e3±9.70e3 | -10.10±27.70 | -12.21±33.65 | -11.06±30.59 | -10.11±27.73 |
| Trimentoring | -321.25±47.40 | -20.28±54.97 | -20.20±55.00 | -20.25±54.98 | -20.23±54.99 | -320.55±47.64 | -20.44±55.24 | -20.36±55.27 | -20.40±55.25 | -20.41±55.24 | -323.08±50.49 | -20.49±55.33 | -20.41±55.35 | -20.45±55.34 | -20.49±55.33 |
| **Task** $f(x^*_{OFF})$ | | | | | | | | GTOPX 6 −121.48±0.38 | | | | | | | |
| ARCOO | -122.10±0.40 | 0.99±0.00 | 0.99±0.00 | 0.99±0.00 | 0.99±0.00 | -122.23±0.38 | 0.99±0.00 | 0.99±0.00 | 0.99±0.00 | 0.99±0.00 | -123.58±3.39 | 0.99±0.00 | 0.99±0.00 | 0.99±0.00 | 0.99±0.00 |
| BO | -187.61±47.13 | 0.43±0.04 | 0.44±0.03 | 0.43±0.03 | 0.43±0.03 | -201.44±41.64 | 0.42±0.05 | 0.43±0.08 | 0.42±0.06 | 0.42±0.05 | -159.63±19.80 | 0.42±0.05 | 0.43±0.06 | 0.43±0.04 | 0.42±0.05 |
| CBAS | -121.48±0.38 | 1.00±0.00 | 1.00±0.00 | 1.00±0.00 | 1.00±0.00 | -121.48±0.38 | 1.00±0.00 | 1.00±0.00 | 1.00±0.00 | 1.00±0.00 | -121.48±0.38 | 1.00±0.00 | 1.00±0.00 | 1.00±0.00 | 1.00±0.00 |
| CCDDEA | -147.79±30.48 | 0.12±0.18 | 0.21±0.31 | 0.15±0.22 | 0.17±0.25 | -566.97±470.66 | -0.81±1.17 | -1.29±1.80 | -0.99±1.41 | -0.92±1.32 | -562.95±483.55 | -1.59±2.16 | -2.59±3.41 | -1.96±2.63 | -1.59±2.17 |
| CMAES | -152.31±4.54 | 0.66±0.06 | 0.66±0.06 | 0.66±0.06 | 0.66±0.06 | - | - | - | - | - | - | - | - | - | - |
| TTDDEA | -109.75±30.30 | 0.75±0.10 | 1.16±0.18 | 0.91±0.12 | 0.98±0.14 | -110.07±33.95 | 0.75±0.10 | 1.18±0.20 | 0.92±0.13 | 0.85±0.11 | -113.08±44.17 | 0.73±0.14 | 1.18±0.26 | 0.90±0.18 | 0.74±0.14 |
| Trimentoring | -142.89±16.60 | -18.68±24.99 | -18.64±25.02 | -18.66±25.00 | -18.65±25.01 | -143.18±16.54 | -18.80±25.10 | -18.76±25.13 | -18.78±25.12 | -18.79±25.11 | -143.91±17.17 | -18.84±25.14 | -18.80±25.17 | -18.82±25.15 | -18.84±25.14 |

Table 22: Overall results for GTOPX unconstrained tasks with 128 solutions and 50th percentile evaluations. In this case, 25% of the values are missing near the worst value and another 25% near the optimal value. Details are the same as Table 5.

| Steps | | | $t = 50$ | | | | | $t = 100$ | | | | | $t = 150$ | | |
|---|---|---|---|---|---|---|---|---|---|---|---|---|---|---|---|
| Task $f(x^*_{OFF})$ | | | | | | | | GTOPX 2 $-195.44_{\pm1.38}$ | | | | | | | |
| Metric | FS | SI | OI | SO | SO$_\omega$ | FS | SI | OI | SO | SO$_\omega$ | FS | SI | OI | SO | SO$_\omega$ |
| ARCOO | $-196.03_{\pm1.45}$ | $1.00_{\pm0.00}$ | $1.00_{\pm0.00}$ | $1.00_{\pm0.00}$ | $1.00_{\pm0.00}$ | $-196.23_{\pm1.41}$ | $1.00_{\pm0.00}$ | $1.00_{\pm0.00}$ | $1.00_{\pm0.00}$ | $1.00_{\pm0.00}$ | $-196.32_{\pm1.50}$ | $0.99_{\pm0.00}$ | $0.99_{\pm0.00}$ | $0.99_{\pm0.00}$ | $0.99_{\pm0.00}$ |
| BO | $-390.14_{\pm78.61}$ | $-0.22_{\pm0.08}$ | $-0.22_{\pm0.08}$ | $-0.22_{\pm0.08}$ | $-0.22_{\pm0.08}$ | $-387.52_{\pm148.75}$ | $-0.19_{\pm0.06}$ | $-0.19_{\pm0.06}$ | $-0.19_{\pm0.06}$ | $-0.19_{\pm0.06}$ | $-459.38_{\pm261.44}$ | $-0.20_{\pm0.06}$ | $-0.21_{\pm0.07}$ | $-0.20_{\pm0.07}$ | $-0.20_{\pm0.06}$ |
| CBAS | $-195.44_{\pm1.38}$ | $1.00_{\pm0.00}$ | $1.00_{\pm0.00}$ | $1.00_{\pm0.00}$ | $1.00_{\pm0.00}$ | $-195.44_{\pm1.38}$ | $1.00_{\pm0.00}$ | $1.00_{\pm0.00}$ | $1.00_{\pm0.00}$ | $1.00_{\pm0.00}$ | $-195.44_{\pm1.38}$ | $1.00_{\pm0.00}$ | $1.00_{\pm0.00}$ | $1.00_{\pm0.00}$ | $1.00_{\pm0.00}$ |
| CCDDEA | $-181.63_{\pm46.41}$ | $0.56_{\pm0.12}$ | $1.02_{\pm0.28}$ | $0.72_{\pm0.17}$ | $0.80_{\pm0.20}$ | $-1.03e3_{\pm871.54}$ | $-0.38_{\pm0.93}$ | $-0.70_{\pm1.70}$ | $-0.49_{\pm1.20}$ | $-0.45_{\pm1.10}$ | $-1.03e3_{\pm871.55}$ | $-1.26_{\pm1.92}$ | $-2.33_{\pm1.49}$ | $-1.64_{\pm2.47}$ | $-1.27_{\pm1.92}$ |
| CMAES | $-142.96_{\pm8.39}$ | $0.95_{\pm0.02}$ | $1.35_{\pm0.09}$ | $1.11_{\pm0.03}$ | $1.18_{\pm0.03}$ | - | - | - | - | - | - | - | - | - | - |
| TTDDEA | $-270.75_{\pm47.88}$ | $0.37_{\pm0.34}$ | $0.44_{\pm0.41}$ | $0.40_{\pm0.37}$ | $0.41_{\pm0.38}$ | $-254.99_{\pm39.60}$ | $0.37_{\pm0.29}$ | $0.44_{\pm0.37}$ | $0.40_{\pm0.32}$ | $0.39_{\pm0.31}$ | $-252.20_{\pm28.39}$ | $0.41_{\pm0.24}$ | $0.48_{\pm0.32}$ | $0.44_{\pm0.27}$ | $0.41_{\pm0.24}$ |
| Trimentoring | $-159.09_{\pm19.13}$ | $-12.38_{\pm35.20}$ | $-12.05_{\pm35.52}$ | $-12.24_{\pm35.25}$ | $-12.18_{\pm35.27}$ | $-160.89_{\pm20.59}$ | $-12.45_{\pm35.38}$ | $-12.13_{\pm35.50}$ | $-12.32_{\pm35.43}$ | $-12.37_{\pm35.41}$ | $-160.20_{\pm18.59}$ | $-12.47_{\pm35.44}$ | $-12.15_{\pm35.56}$ | $-12.34_{\pm35.49}$ | $-12.47_{\pm35.44}$ |
| Task $f(x^*_{OFF})$ | | | | | | | | GTOPX 3 $-151.85_{\pm0.66}$ | | | | | | | |
| ARCOO | $-152.59_{\pm0.70}$ | $0.99_{\pm0.00}$ | $0.99_{\pm0.00}$ | $0.99_{\pm0.00}$ | $0.99_{\pm0.00}$ | $-152.79_{\pm0.85}$ | $0.99_{\pm0.00}$ | $0.99_{\pm0.00}$ | $0.99_{\pm0.00}$ | $0.99_{\pm0.00}$ | $-152.98_{\pm0.92}$ | $0.99_{\pm0.00}$ | $0.99_{\pm0.00}$ | $0.99_{\pm0.00}$ | $0.99_{\pm0.00}$ |
| BO | $-353.05_{\pm88.80}$ | $-0.84_{\pm0.18}$ | $-0.85_{\pm0.18}$ | $-0.84_{\pm0.18}$ | $-0.85_{\pm0.18}$ | $-401.86_{\pm177.88}$ | $-0.83_{\pm0.14}$ | $-0.84_{\pm0.14}$ | $-0.83_{\pm0.14}$ | $-0.83_{\pm0.14}$ | $-406.74_{\pm180.22}$ | $-0.85_{\pm0.11}$ | $-0.86_{\pm0.11}$ | $-0.85_{\pm0.11}$ | $-0.85_{\pm0.11}$ |
| CBAS | $-151.85_{\pm0.66}$ | $1.00_{\pm0.00}$ | $1.00_{\pm0.00}$ | $1.00_{\pm0.00}$ | $1.00_{\pm0.00}$ | $-151.85_{\pm0.66}$ | $1.00_{\pm0.00}$ | $1.00_{\pm0.00}$ | $1.00_{\pm0.00}$ | $1.00_{\pm0.00}$ | $-151.85_{\pm0.66}$ | $1.00_{\pm0.00}$ | $1.00_{\pm0.00}$ | $1.00_{\pm0.00}$ | $1.00_{\pm0.00}$ |
| CCDDEA | $-5.99e4_{\pm1.86e5}$ | $-6.72_{\pm9.56}$ | $-11.35_{\pm16.15}$ | $-8.44_{\pm12.01}$ | $-9.24_{\pm13.15}$ | $-215.02_{\pm43.44}$ | $-5.58_{\pm11.04}$ | $-9.47_{\pm18.65}$ | $-7.03_{\pm13.87}$ | $-6.48_{\pm12.80}$ | $-215.07_{\pm43.41}$ | $-3.63_{\pm7.43}$ | $-6.16_{\pm12.56}$ | $-4.57_{\pm9.34}$ | $-3.64_{\pm7.45}$ |
| CMAES | $-129.66_{\pm7.15}$ | $0.95_{\pm0.02}$ | $1.18_{\pm0.04}$ | $1.05_{\pm0.02}$ | $1.09_{\pm0.03}$ | - | - | - | - | - | - | - | - | - | - |
| TTDDEA | $-187.44_{\pm105.64}$ | $0.36_{\pm0.48}$ | $0.67_{\pm0.82}$ | $0.47_{\pm0.61}$ | $0.53_{\pm0.66}$ | $-2.98e3_{\pm4.15e3}$ | $-0.88_{\pm1.43}$ | $-1.19_{\pm2.00}$ | $-1.01_{\pm1.65}$ | $-0.97_{\pm1.57}$ | $-4.23e6_{\pm1.07e7}$ | $-931.83_{\pm2.25e3}$ | $-1.22e3_{\pm2.88e3}$ | $-1.06e3_{\pm2.53e3}$ | $-933.27_{\pm2.26e3}$ |
| Trimentoring | $-150.41_{\pm4.20}$ | $-57.98_{\pm59.39}$ | $-57.92_{\pm59.45}$ | $-57.95_{\pm59.42}$ | $-57.94_{\pm59.43}$ | $-150.44_{\pm4.23}$ | $-58.29_{\pm59.70}$ | $-58.23_{\pm59.76}$ | $-58.26_{\pm59.73}$ | $-58.27_{\pm59.72}$ | $-149.05_{\pm7.51}$ | $-58.39_{\pm59.80}$ | $-58.33_{\pm59.86}$ | $-58.36_{\pm59.83}$ | $-58.39_{\pm59.80}$ |
| Task $f(x^*_{OFF})$ | | | | | | | | GTOPX 4 $-112.11_{\pm0.33}$ | | | | | | | |
| ARCOO | $-216.88_{\pm1.09}$ | $0.99_{\pm0.00}$ | $0.99_{\pm0.00}$ | $0.99_{\pm0.00}$ | $0.99_{\pm0.00}$ | $-217.34_{\pm1.33}$ | $0.99_{\pm0.00}$ | $0.99_{\pm0.00}$ | $0.99_{\pm0.00}$ | $0.99_{\pm0.00}$ | $-217.54_{\pm1.34}$ | $0.99_{\pm0.00}$ | $0.99_{\pm0.00}$ | $0.99_{\pm0.00}$ | $0.99_{\pm0.00}$ |
| BO | $-698.38_{\pm168.67}$ | $-1.14_{\pm0.28}$ | $-1.14_{\pm0.28}$ | $-1.14_{\pm0.28}$ | $-1.14_{\pm0.28}$ | $-504.00_{\pm151.96}$ | $-1.18_{\pm0.14}$ | $-1.18_{\pm0.14}$ | $-1.18_{\pm0.14}$ | $-1.18_{\pm0.14}$ | $-420.32_{\pm117.19}$ | $-1.19_{\pm0.12}$ | $-1.19_{\pm0.12}$ | $-1.19_{\pm0.12}$ | $-1.19_{\pm0.12}$ |
| CBAS | $-215.74_{\pm1.18}$ | $1.00_{\pm0.00}$ | $1.00_{\pm0.00}$ | $1.00_{\pm0.00}$ | $1.00_{\pm0.00}$ | $-215.74_{\pm1.18}$ | $1.00_{\pm0.00}$ | $1.00_{\pm0.00}$ | $1.00_{\pm0.00}$ | $1.00_{\pm0.00}$ | $-215.74_{\pm1.18}$ | $1.00_{\pm0.00}$ | $1.00_{\pm0.00}$ | $1.00_{\pm0.00}$ | $1.00_{\pm0.00}$ |
| CCDDEA | $-425.49_{\pm340.04}$ | $-7.49_{\pm13.66}$ | $-11.81_{\pm22.71}$ | $-9.13_{\pm17.06}$ | $-9.88_{\pm18.62}$ | $-1.95e5_{\pm2.94e5}$ | $-436.30_{\pm899.29}$ | $-594.08_{\pm782.58}$ | $-501.16_{\pm676.76}$ | $-477.81_{\pm649.26}$ | $-9.80e4_{\pm1.61e5}$ | $-614.22_{\pm864.73}$ | $-819.71_{\pm1.10e3}$ | $-699.98_{\pm967.88}$ | $-615.21_{\pm865.95}$ |
| CMAES | $-317.51_{\pm14.09}$ | $0.33_{\pm0.10}$ | $0.33_{\pm0.10}$ | $0.33_{\pm0.10}$ | $0.33_{\pm0.10}$ | - | - | - | - | - | - | - | - | - | - |
| TTDDEA | $-570.92_{\pm729.80}$ | $-1.16_{\pm3.70}$ | $-1.03_{\pm3.77}$ | $-1.11_{\pm3.75}$ | $-1.08_{\pm3.74}$ | $-1.75e3_{\pm3.97e3}$ | $-3.00_{\pm8.69}$ | $-2.88_{\pm8.74}$ | $-2.96_{\pm8.71}$ | $-2.97_{\pm8.70}$ | $-542.45_{\pm437.79}$ | $-3.98_{\pm11.32}$ | $-3.85_{\pm11.37}$ | $-3.93_{\pm11.34}$ | $-3.98_{\pm11.32}$ |
| Trimentoring | $-300.76_{\pm66.10}$ | $-29.18_{\pm51.54}$ | $-29.16_{\pm51.55}$ | $-29.17_{\pm51.55}$ | $-29.17_{\pm51.55}$ | $-300.26_{\pm65.53}$ | $-29.39_{\pm51.78}$ | $-29.38_{\pm51.79}$ | $-29.39_{\pm51.78}$ | $-29.39_{\pm51.78}$ | $-299.54_{\pm64.75}$ | $-29.46_{\pm51.86}$ | $-29.45_{\pm51.86}$ | $-29.45_{\pm51.86}$ | $-29.46_{\pm51.86}$ |
| Task $f(x^*_{OFF})$ | | | | | | | | GTOPX 6 $-215.74_{\pm1.18}$ | | | | | | | |
| ARCOO | $-112.71_{\pm0.42}$ | $0.99_{\pm0.00}$ | $0.99_{\pm0.00}$ | $0.99_{\pm0.00}$ | $0.99_{\pm0.00}$ | $-116.53_{\pm10.20}$ | $0.99_{\pm0.01}$ | $0.99_{\pm0.01}$ | $0.99_{\pm0.01}$ | $0.99_{\pm0.01}$ | $-114.63_{\pm4.60}$ | $0.97_{\pm0.04}$ | $0.97_{\pm0.04}$ | $0.97_{\pm0.04}$ | $0.97_{\pm0.04}$ |
| BO | $-170.14_{\pm30.51}$ | $0.23_{\pm0.06}$ | $0.23_{\pm0.06}$ | $0.23_{\pm0.06}$ | $0.23_{\pm0.06}$ | $-187.78_{\pm38.08}$ | $0.23_{\pm0.05}$ | $0.23_{\pm0.05}$ | $0.23_{\pm0.05}$ | $0.23_{\pm0.05}$ | $-174.63_{\pm21.25}$ | $0.23_{\pm0.04}$ | $0.23_{\pm0.04}$ | $0.23_{\pm0.04}$ | $0.23_{\pm0.04}$ |
| CBAS | $-112.11_{\pm0.33}$ | $1.00_{\pm0.00}$ | $1.00_{\pm0.00}$ | $1.00_{\pm0.00}$ | $1.00_{\pm0.00}$ | $-112.11_{\pm0.33}$ | $1.00_{\pm0.00}$ | $1.00_{\pm0.00}$ | $1.00_{\pm0.00}$ | $1.00_{\pm0.00}$ | $-112.11_{\pm0.33}$ | $1.00_{\pm0.00}$ | $1.00_{\pm0.00}$ | $1.00_{\pm0.00}$ | $1.00_{\pm0.00}$ |
| CCDDEA | $-173.75_{\pm66.02}$ | $0.12_{\pm0.30}$ | $0.19_{\pm0.43}$ | $0.14_{\pm0.35}$ | $0.16_{\pm0.37}$ | $-3.54e3_{\pm3.35e3}$ | $-9.03_{\pm15.48}$ | $-13.73_{\pm22.33}$ | $-10.86_{\pm18.21}$ | $-10.18_{\pm17.21}$ | $-3.53e3_{\pm3.36e3}$ | $-18.08_{\pm30.96}$ | $-27.54_{\pm44.77}$ | $-21.75_{\pm36.45}$ | $-18.12_{\pm31.02}$ |
| CMAES | $-155.22_{\pm5.00}$ | $0.49_{\pm0.06}$ | $0.49_{\pm0.06}$ | $0.49_{\pm0.06}$ | $0.49_{\pm0.06}$ | - | - | - | - | - | - | - | - | - | - |
| TTDDEA | $-147.47_{\pm61.86}$ | $0.22_{\pm0.50}$ | $0.37_{\pm0.75}$ | $0.28_{\pm0.60}$ | $0.30_{\pm0.64}$ | $-161.40_{\pm70.06}$ | $0.20_{\pm0.59}$ | $0.35_{\pm0.82}$ | $0.25_{\pm0.68}$ | $0.23_{\pm0.64}$ | $-179.32_{\pm87.52}$ | $0.17_{\pm0.62}$ | $0.30_{\pm0.87}$ | $0.22_{\pm0.71}$ | $0.17_{\pm0.62}$ |
| Trimentoring | $-229.02_{\pm258.01}$ | $-14.11_{\pm20.35}$ | $-14.11_{\pm20.35}$ | $-14.11_{\pm20.35}$ | $-14.11_{\pm20.35}$ | $-159.48_{\pm65.48}$ | $-13.38_{\pm20.68}$ | $-13.38_{\pm20.68}$ | $-13.38_{\pm20.68}$ | $-13.38_{\pm20.68}$ | $-148.71_{\pm48.07}$ | $-12.96_{\pm20.88}$ | $-12.96_{\pm20.88}$ | $-12.96_{\pm20.88}$ | $-12.96_{\pm20.88}$ |

Table 23: Overall results for GTOPX unconstrained tasks with 128 solutions and 50th percentile evaluations. In this case, 30% of the values are missing near the worst value and another 20% near the optimal value. Details are the same as Table 5.

| Steps | | | $t = 50$ | | | | | $t = 100$ | | | | | $t = 150$ | | |
|---|---|---|---|---|---|---|---|---|---|---|---|---|---|---|---|
| Task $f(x^*_{OFF})$ | | | | | | | | GTOPX 2 $-175.10_{\pm0.75}$ | | | | | | | |
| Metric | FS | SI | OI | SO | SO$_\omega$ | FS | SI | OI | SO | SO$_\omega$ | FS | SI | OI | SO | SO$_\omega$ |
| ARCOO | $-175.84_{\pm0.95}$ | $0.99_{\pm0.00}$ | $0.99_{\pm0.00}$ | $0.99_{\pm0.00}$ | $0.99_{\pm0.00}$ | $-176.07_{\pm1.05}$ | $0.99_{\pm0.00}$ | $0.99_{\pm0.00}$ | $0.99_{\pm0.00}$ | $0.99_{\pm0.00}$ | $-176.22_{\pm1.03}$ | $0.99_{\pm0.00}$ | $0.99_{\pm0.00}$ | $0.99_{\pm0.00}$ | $0.99_{\pm0.00}$ |
| BO | $-429.01_{\pm91.54}$ | $-0.56_{\pm0.13}$ | $-0.56_{\pm0.13}$ | $-0.56_{\pm0.13}$ | $-0.56_{\pm0.13}$ | $-356.29_{\pm92.87}$ | $-0.58_{\pm0.12}$ | $-0.58_{\pm0.12}$ | $-0.58_{\pm0.12}$ | $-0.58_{\pm0.12}$ | $-385.35_{\pm158.06}$ | $-0.59_{\pm0.09}$ | $-0.59_{\pm0.09}$ | $-0.59_{\pm0.09}$ | $-0.59_{\pm0.09}$ |
| CBAS | $-175.10_{\pm0.75}$ | $1.00_{\pm0.00}$ | $1.00_{\pm0.00}$ | $1.00_{\pm0.00}$ | $1.00_{\pm0.00}$ | $-175.10_{\pm0.75}$ | $1.00_{\pm0.00}$ | $1.00_{\pm0.00}$ | $1.00_{\pm0.00}$ | $1.00_{\pm0.00}$ | $-175.10_{\pm0.75}$ | $1.00_{\pm0.00}$ | $1.00_{\pm0.00}$ | $1.00_{\pm0.00}$ | $1.00_{\pm0.00}$ |
| CCDDEA | $-176.33_{\pm28.82}$ | $0.20_{\pm0.77}$ | $0.32_{\pm0.66}$ | $0.24_{\pm0.34}$ | $0.26_{\pm0.37}$ | $-1.31e3_{\pm1.81e3}$ | $-1.35_{\pm2.83}$ | $-2.14_{\pm4.45}$ | $-1.65_{\pm3.46}$ | $-1.54_{\pm3.23}$ | $-1.55e3_{\pm1.5e3}$ | $-3.12_{\pm5.89}$ | $-4.97_{\pm9.23}$ | $-3.83_{\pm7.19}$ | $-3.13_{\pm5.91}$ |
| CMAES | $-139.70_{\pm4.28}$ | $0.94_{\pm0.01}$ | $1.26_{\pm0.02}$ | $1.08_{\pm0.01}$ | $1.14_{\pm0.01}$ | - | - | - | - | - | - | - | - | - | - |
| TTDDEA | $-282.14_{\pm53.79}$ | $0.09_{\pm0.60}$ | $0.12_{\pm0.61}$ | $0.10_{\pm0.61}$ | $0.11_{\pm0.61}$ | $-266.25_{\pm35.22}$ | $0.10_{\pm0.47}$ | $0.11_{\pm0.47}$ | $0.11_{\pm0.47}$ | $0.11_{\pm0.47}$ | $-269.60_{\pm30.93}$ | $0.11_{\pm0.39}$ | $0.12_{\pm0.39}$ | $0.12_{\pm0.39}$ | $0.11_{\pm0.39}$ |
| Trimentoring | $-4.21e3_{\pm1.07e4}$ | $-60.15_{\pm123.48}$ | $-59.97_{\pm125.99}$ | $-60.07_{\pm129.54}$ | $-60.04_{\pm125.56}$ | $-4.31e3_{\pm1.10e4}$ | $-60.62_{\pm126.56}$ | $-60.44_{\pm126.65}$ | $-60.55_{\pm126.60}$ | $-60.58_{\pm126.59}$ | $-5.25e3_{\pm1.35e4}$ | $-64.03_{\pm135.16}$ | $-63.85_{\pm135.25}$ | $-63.95_{\pm135.19}$ | $-64.03_{\pm135.16}$ |
| Task $f(x^*_{OFF})$ | | | | | | | | GTOPX 3 $-134.83_{\pm0.56}$ | | | | | | | |
| ARCOO | $-135.75_{\pm0.69}$ | $0.99_{\pm0.00}$ | $0.99_{\pm0.00}$ | $0.99_{\pm0.00}$ | $0.99_{\pm0.00}$ | $-136.12_{\pm0.65}$ | $0.99_{\pm0.00}$ | $0.99_{\pm0.00}$ | $0.99_{\pm0.00}$ | $0.99_{\pm0.00}$ | $-136.23_{\pm0.64}$ | $0.99_{\pm0.00}$ | $0.99_{\pm0.00}$ | $0.99_{\pm0.00}$ | $0.99_{\pm0.00}$ |
| BO | $-380.20_{\pm80.18}$ | $-1.22_{\pm0.16}$ | $-1.22_{\pm0.16}$ | $-1.22_{\pm0.16}$ | $-1.22_{\pm0.16}$ | $-332.79_{\pm46.61}$ | $-1.28_{\pm0.17}$ | $-1.28_{\pm0.17}$ | $-1.28_{\pm0.17}$ | $-1.28_{\pm0.17}$ | $-305.41_{\pm106.02}$ | $-1.24_{\pm0.17}$ | $-1.24_{\pm0.17}$ | $-1.24_{\pm0.17}$ | $-1.24_{\pm0.17}$ |
| CBAS | $-134.83_{\pm0.56}$ | $1.00_{\pm0.00}$ | $1.00_{\pm0.00}$ | $1.00_{\pm0.00}$ | $1.00_{\pm0.00}$ | $-134.83_{\pm0.56}$ | $1.00_{\pm0.00}$ | $1.00_{\pm0.00}$ | $1.00_{\pm0.00}$ | $1.00_{\pm0.00}$ | $-134.83_{\pm0.56}$ | $1.00_{\pm0.00}$ | $1.00_{\pm0.00}$ | $1.00_{\pm0.00}$ | $1.00_{\pm0.00}$ |
| CCDDEA | $-154.74_{\pm46.42}$ | $-5.80_{\pm11.25}$ | $-10.99_{\pm21.97}$ | $-7.59_{\pm14.79}$ | $-8.48_{\pm16.54}$ | $-323.78_{\pm296.76}$ | $-3.01_{\pm5.96}$ | $-5.73_{\pm11.42}$ | $-3.95_{\pm7.83}$ | $-3.58_{\pm7.11}$ | $-323.66_{\pm296.79}$ | $-2.26_{\pm4.63}$ | $-4.30_{\pm8.86}$ | $-2.96_{\pm6.08}$ | $-2.27_{\pm4.64}$ |
| CMAES | $-131.81_{\pm4.27}$ | $0.93_{\pm0.01}$ | $1.03_{\pm0.04}$ | $0.98_{\pm0.02}$ | $0.99_{\pm0.02}$ | - | - | - | - | - | - | - | - | - | - |
| TTDDEA | $-212.65_{\pm100.96}$ | $-0.17_{\pm1.52}$ | $0.08_{\pm1.67}$ | $-0.03_{\pm1.60}$ | $-0.03_{\pm1.60}$ | $-6.67e3_{\pm1.10e4}$ | $-5.80_{\pm10.25}$ | $-6.23_{\pm10.61}$ | $-5.98_{\pm10.37}$ | $-5.92_{\pm10.32}$ | $-4.18e6_{\pm1.06e7}$ | $-5.09e3_{\pm1.30e4}$ | $-5.14e3_{\pm1.30e4}$ | $-5.11e3_{\pm1.30e4}$ | $-5.09e3_{\pm1.30e4}$ |
| Trimentoring | $-138.26_{\pm3.46}$ | $0.86_{\pm0.07}$ | $0.99_{\pm0.03}$ | $0.92_{\pm0.04}$ | $0.94_{\pm0.03}$ | $-140.50_{\pm3.77}$ | $0.84_{\pm0.08}$ | $0.97_{\pm0.04}$ | $0.90_{\pm0.06}$ | $0.89_{\pm0.06}$ | $-139.81_{\pm4.58}$ | $0.83_{\pm0.09}$ | $0.96_{\pm0.04}$ | $0.89_{\pm0.06}$ | $0.83_{\pm0.08}$ |
| Task $f(x^*_{OFF})$ | | | | | | | | GTOPX 4 $-194.03_{\pm0.88}$ | | | | | | | |
| ARCOO | $-195.14_{\pm0.82}$ | $0.99_{\pm0.00}$ | $0.99_{\pm0.00}$ | $0.99_{\pm0.00}$ | $0.99_{\pm0.00}$ | $-195.45_{\pm0.70}$ | $0.99_{\pm0.00}$ | $0.99_{\pm0.00}$ | $0.99_{\pm0.00}$ | $0.99_{\pm0.00}$ | $-198.95_{\pm8.94}$ | $0.99_{\pm0.01}$ | $0.99_{\pm0.01}$ | $0.99_{\pm0.01}$ | $0.99_{\pm0.01}$ |
| BO | $-580.82_{\pm193.20}$ | $-1.78_{\pm0.32}$ | $-1.78_{\pm0.32}$ | $-1.78_{\pm0.32}$ | $-1.78_{\pm0.32}$ | $-491.92_{\pm190.12}$ | $-1.69_{\pm0.16}$ | $-1.69_{\pm0.16}$ | $-1.69_{\pm0.16}$ | $-1.69_{\pm0.16}$ | $-496.29_{\pm313.60}$ | $-1.69_{\pm0.16}$ | $-1.69_{\pm0.16}$ | $-1.69_{\pm0.16}$ | $-1.69_{\pm0.16}$ |
| CBAS | $-194.03_{\pm0.88}$ | $1.00_{\pm0.00}$ | $1.00_{\pm0.00}$ | $1.00_{\pm0.00}$ | $1.00_{\pm0.00}$ | $-194.03_{\pm0.88}$ | $1.00_{\pm0.00}$ | $1.00_{\pm0.00}$ | $1.00_{\pm0.00}$ | $1.00_{\pm0.00}$ | $-194.03_{\pm0.88}$ | $1.00_{\pm0.00}$ | $1.00_{\pm0.00}$ | $1.00_{\pm0.00}$ | $1.00_{\pm0.00}$ |
| CCDDEA | $-460.90_{\pm447.76}$ | $-47.90_{\pm107.98}$ | $-60.44_{\pm130.16}$ | $-53.30_{\pm118.06}$ | $-55.47_{\pm121.89}$ | $-2.04e5_{\pm4.07e5}$ | $-420.85_{\pm767.60}$ | $-709.36_{\pm1.41e3}$ | $-524.88_{\pm991.15}$ | $-485.38_{\pm905.17}$ | $-3.22e6_{\pm7.84e6}$ | $-4.72e3_{\pm1.04e4}$ | $-6.14e3_{\pm1.26e4}$ | $-5.31e3_{\pm1.14e4}$ | $-4.73e3_{\pm1.04e4}$ |
| CMAES | $-322.45_{\pm13.63}$ | $-0.02_{\pm0.09}$ | $-0.02_{\pm0.09}$ | $-0.02_{\pm0.09}$ | $-0.02_{\pm0.09}$ | - | - | - | - | - | - | - | - | - | - |
| TTDDEA | $-278.04_{\pm63.15}$ | $0.10_{\pm0.35}$ | $0.14_{\pm0.47}$ | $0.12_{\pm0.40}$ | $0.12_{\pm0.42}$ | $-296.84_{\pm57.66}$ | $0.13_{\pm0.32}$ | $0.17_{\pm0.42}$ | $0.15_{\pm0.36}$ | $0.14_{\pm0.34}$ | $-285.40_{\pm64.27}$ | $0.14_{\pm0.28}$ | $0.18_{\pm0.37}$ | $0.15_{\pm0.32}$ | $0.14_{\pm0.28}$ |
| Trimentoring | $-325.30_{\pm75.26}$ | $-25.89_{\pm44.56}$ | $-25.90_{\pm44.55}$ | $-25.89_{\pm44.55}$ | $-25.90_{\pm44.55}$ | $-328.32_{\pm77.34}$ | $-26.09_{\pm44.76}$ | $-26.10_{\pm44.75}$ | $-26.09_{\pm44.75}$ | $-26.09_{\pm44.75}$ | $-329.52_{\pm77.63}$ | $-26.15_{\pm44.82}$ | $-26.17_{\pm44.81}$ | $-26.16_{\pm44.82}$ | $-26.15_{\pm44.82}$ |
| Task $f(x^*_{OFF})$ | | | | | | | | GTOPX 6 $-102.96_{\pm0.29}$ | | | | | | | |
| ARCOO | $-103.69_{\pm0.40}$ | $0.99_{\pm0.00}$ | $0.99_{\pm0.00}$ | $0.99_{\pm0.00}$ | $0.99_{\pm0.00}$ | $-103.79_{\pm0.39}$ | $0.99_{\pm0.00}$ | $0.99_{\pm0.00}$ | $0.99_{\pm0.00}$ | $0.99_{\pm0.00}$ | $-109.16_{\pm11.83}$ | $0.98_{\pm0.03}$ | $0.98_{\pm0.03}$ | $0.98_{\pm0.03}$ | $0.98_{\pm0.03}$ |
| BO | $-176.47_{\pm11.71}$ | $0.02_{\pm0.06}$ | $0.02_{\pm0.06}$ | $0.02_{\pm0.06}$ | $0.02_{\pm0.06}$ | $-184.56_{\pm29.11}$ | $0.02_{\pm0.06}$ | $0.02_{\pm0.06}$ | $0.02_{\pm0.06}$ | $0.02_{\pm0.06}$ | $-178.53_{\pm45.07}$ | $0.02_{\pm0.05}$ | $0.02_{\pm0.05}$ | $0.02_{\pm0.05}$ | $0.02_{\pm0.05}$ |
| CBAS | $-102.96_{\pm0.29}$ | $1.00_{\pm0.00}$ | $1.00_{\pm0.00}$ | $1.00_{\pm0.00}$ | $1.00_{\pm0.00}$ | $-102.96_{\pm0.29}$ | $1.00_{\pm0.00}$ | $1.00_{\pm0.00}$ | $1.00_{\pm0.00}$ | $1.00_{\pm0.00}$ | $-102.96_{\pm0.29}$ | $1.00_{\pm0.00}$ | $1.00_{\pm0.00}$ | $1.00_{\pm0.00}$ | $1.00_{\pm0.00}$ |
| CCDDEA | $-152.76_{\pm46.65}$ | $0.10_{\pm0.21}$ | $0.16_{\pm0.33}$ | $0.12_{\pm0.26}$ | $0.13_{\pm0.28}$ | $-1.11e3_{\pm1.69e3}$ | $-2.35_{\pm3.82}$ | $-3.85_{\pm6.28}$ | $-2.92_{\pm4.75}$ | $-2.71_{\pm4.40}$ | $-1.37e3_{\pm1.71e3}$ | $-5.28_{\pm7.77}$ | $-8.65_{\pm12.82}$ | $-6.56_{\pm9.67}$ | $-5.30_{\pm7.79}$ |
| CMAES | $-155.62_{\pm4.46}$ | $0.28_{\pm0.07}$ | $0.28_{\pm0.07}$ | $0.28_{\pm0.07}$ | $0.28_{\pm0.07}$ | - | - | - | - | - | - | - | - | - | - |
| TTDDEA | $-156.03_{\pm78.68}$ | $0.02_{\pm0.66}$ | $0.06_{\pm0.92}$ | $0.03_{\pm0.77}$ | $0.04_{\pm0.81}$ | $-122.23_{\pm40.35}$ | $0.17_{\pm0.59}$ | $0.25_{\pm0.76}$ | $0.20_{\pm0.66}$ | $0.19_{\pm0.64}$ | $-118.43_{\pm37.83}$ | $0.29_{\pm0.41}$ | $0.42_{\pm0.57}$ | $0.35_{\pm0.47}$ | $0.29_{\pm0.41}$ |
| Trimentoring | $-3.18e4_{\pm8.36e4}$ | $-418.21_{\pm1.03e3}$ | $-418.51_{\pm1.03e3}$ | $-418.36_{\pm1.03e3}$ | $-418.41_{\pm1.03e3}$ | $-3.13e4_{\pm8.25e4}$ | $-467.85_{\pm1.18e3}$ | $-468.00_{\pm1.18e3}$ | $-467.92_{\pm1.18e3}$ | $-467.90_{\pm1.18e3}$ | $-3.03e4_{\pm7.95e4}$ | $-476.66_{\pm1.21e3}$ | $-476.76_{\pm1.21e3}$ | $-476.71_{\pm1.21e3}$ | $-476.66_{\pm1.21e3}$ |

Table 24: Overall results for GTOPX unconstrained tasks with 128 solutions and 50th percentile evaluations. In this case, 40% of the values are missing near the worst value and another 10% near the optimal value. Details are the same as Table 5.

| Steps | | t = 50 | | | | | t = 100 | | | | | t = 150 | | | |
|---|---|---|---|---|---|---|---|---|---|---|---|---|---|---|---|
| Task $f(x^*_{OFF})$ | | | | | | | GTOPX 2 -136.28±0.39 | | | | | | | | |
| Metric | FS | SI | OI | SO | SO$_\omega$ | FS | SI | OI | SO | SO$_\omega$ | FS | SI | OI | SO | SO$_\omega$ |
| ARCOO | -137.50±0.57 | 0.99±0.00 | 0.99±0.00 | 0.99±0.00 | 0.99±0.00 | -137.69±0.55 | 0.98±0.00 | 0.98±0.00 | 0.98±0.00 | 0.98±0.00 | -137.83±0.57 | 0.98±0.00 | 0.98±0.00 | 0.98±0.00 | 0.98±0.00 |
| BO | -345.66±60.44 | -1.76±0.21 | -1.76±0.21 | -1.76±0.21 | -1.76±0.21 | -379.80±80.73 | -1.81±0.22 | -1.81±0.22 | -1.81±0.22 | -1.81±0.22 | -357.23±47.74 | -1.80±0.20 | -1.80±0.20 | -1.80±0.20 | -1.80±0.20 |
| CBAS | -136.28±0.39 | 1.00±0.00 | 1.00±0.00 | 1.00±0.00 | 1.00±0.00 | -136.28±0.39 | 1.00±0.00 | 1.00±0.00 | 1.00±0.00 | 1.00±0.00 | -136.28±0.39 | 1.00±0.00 | 1.00±0.00 | 1.00±0.00 | 1.00±0.00 |
| CCDDEA | -217.72±173.32 | 0.01±0.91 | 0.08±1.41 | 0.03±1.20 | 0.04±1.19 | -1.41e4±3.03e4 | -25.20±55.61 | -43.53±195.87 | -31.92±70.39 | -29.38±64.80 | -6.01e4±1.50e5 | -113.54±274.86 | -196.25±473.71 | -143.84±347.88 | -113.86±275.62 |
| CMAES | -138.41±5.04 | 0.92±0.02 | 0.99±0.04 | 0.96±0.03 | 0.97±0.03 | - | - | - | - | - | - | - | - | - | - |
| TTDDEA | -355.53±77.82 | -2.11±1.29 | -2.11±1.29 | -2.11±1.29 | -2.11±1.29 | -336.45±71.01 | -2.26±1.18 | -2.27±1.17 | -2.26±1.17 | -2.26±1.18 | -336.04±78.65 | -2.26±1.20 | -2.27±1.19 | -2.27±1.39 | -2.26±1.20 |
| Trimentoring | -137.42±1.14 | -41.38±54.58 | -41.34±54.62 | -41.36±54.60 | -41.35±58.13 | -137.03±0.96 | -41.60±54.86 | -41.56±54.88 | -41.58±54.88 | -41.59±54.87 | -138.22±2.67 | -41.68±54.95 | -41.64±54.98 | -41.66±54.97 | -41.68±54.95 |
| Task $f(x^*_{OFF})$ | | | | | | | GTOPX 3 -102.69±0.49 | | | | | | | | |
| ARCOO | -104.43±0.48 | 0.98±0.00 | 0.98±0.00 | 0.98±0.00 | 0.98±0.00 | -104.65±0.39 | 0.97±0.01 | 0.97±0.00 | 0.97±0.01 | 0.97±0.01 | -108.61±9.97 | 0.97±0.01 | 0.97±0.01 | 0.97±0.01 | 0.97±0.01 |
| BO | -277.86±31.54 | -2.59±0.35 | -2.59±0.35 | -2.59±0.35 | -2.59±0.35 | -397.88±109.28 | -2.59±0.28 | -2.59±0.28 | -2.59±0.28 | -2.59±0.28 | -323.62±108.69 | -2.59±0.31 | -2.59±0.31 | -2.59±0.31 | -2.59±0.31 |
| CBAS | -102.69±0.49 | 1.00±0.00 | 1.00±0.00 | 1.00±0.00 | 1.00±0.00 | -102.69±0.49 | 1.00±0.00 | 1.00±0.00 | 1.00±0.00 | 1.00±0.00 | -102.69±0.49 | 1.00±0.00 | 1.00±0.00 | 1.00±0.00 | 1.00±0.00 |
| CCDDEA | -129.19±34.30 | -32.04±79.39 | -55.30±137.04 | -40.57±100.54 | -44.58±110.48 | -2.51e3±4.37e3 | -19.72±38.49 | -37.61±79.27 | -25.86±50.93 | -23.49±46.10 | -2.57e3±4.34e3 | -19.91±32.21 | -37.17±61.46 | -25.91±42.23 | -19.97±32.31 |
| CMAES | -127.94±6.74 | 0.62±0.13 | 0.62±0.13 | 0.62±0.13 | 0.62±0.13 | - | - | - | - | - | - | - | - | - | - |
| TTDDEA | -146.12±94.61 | -0.48±2.12 | -0.29±2.22 | -0.40±2.16 | -0.37±2.18 | -1.54e5±4.08e5 | -468.44±1.24e3 | -468.26±1.24e3 | -468.37±1.24e3 | -468.39±1.24e3 | -3.32e5±5.54e5 | -2.55e3±5.58e3 | -2.56e3±5.57e3 | -2.56e3±5.57e3 | -2.55e3±5.58e3 |
| Trimentoring | -111.37±12.66 | -74.21±58.13 | -74.21±58.14 | -74.21±58.13 | -74.21±58.13 | -114.18±17.57 | -74.62±58.42 | -74.62±58.42 | -74.62±58.42 | -74.62±58.42 | -115.52±19.85 | -74.76±58.50 | -74.76±58.50 | -74.76±58.50 | -74.76±58.50 |
| Task $f(x^*_{OFF})$ | | | | | | | GTOPX 4 -153.62±0.60 | | | | | | | | |
| ARCOO | -155.27±0.65 | 0.98±0.00 | 0.98±0.00 | 0.98±0.00 | 0.98±0.00 | -156.83±3.05 | 0.98±0.01 | 0.98±0.00 | 0.98±0.01 | 0.98±0.01 | -157.29±4.44 | 0.98±0.01 | 0.98±0.01 | 0.98±0.01 | 0.98±0.01 |
| BO | -566.28±228.88 | -3.36±0.49 | -3.36±0.49 | -3.36±0.49 | -3.36±0.49 | -475.59±226.20 | -3.28±0.47 | -3.28±0.47 | -3.28±0.47 | -3.28±0.47 | -517.40±159.92 | -3.35±0.40 | -3.35±0.40 | -3.35±0.40 | -3.35±0.40 |
| CBAS | -153.62±0.60 | 1.00±0.00 | 1.00±0.00 | 1.00±0.00 | 1.00±0.00 | -153.62±0.60 | 1.00±0.00 | 1.00±0.00 | 1.00±0.00 | 1.00±0.00 | -153.62±0.60 | 1.00±0.00 | 1.00±0.00 | 1.00±0.00 | 1.00±0.00 |
| CCDDEA | -1.12e3±1.99e3 | -730.67±1.85e3 | -1.06e3±2.67e3 | -863.38±2.19e3 | -919.91±2.23e3 | -6.24e5±1.52e6 | -2.02e3±4.74e3 | -2.85e3±6.84e3 | -2.36e3±5.60e3 | -2.24e3±5.29e3 | -6.30e5±1.51e6 | -3.17e3±7.37e3 | -4.44e3±1.06e4 | -3.69e3±8.76e3 | -3.17e3±7.39e3 |
| CMAES | -328.15±9.85 | -1.13±0.24 | -1.13±0.24 | -1.13±0.24 | -1.13±0.24 | - | - | - | - | - | - | - | - | - | - |
| TTDDEA | -274.28±97.04 | -1.04±1.24 | -1.05±1.28 | -1.05±1.26 | -1.05±1.26 | -333.26±114.83 | -1.09±1.39 | -1.07±1.43 | -1.09±1.41 | -1.09±1.40 | -348.71±116.17 | -1.28±1.45 | -1.27±1.50 | -1.27±1.47 | -1.28±1.45 |
| Trimentoring | -270.14±115.53 | -42.81±54.63 | -42.81±54.63 | -42.81±54.63 | -42.81±54.63 | -266.97±112.67 | -43.09±54.88 | -43.09±54.88 | -43.09±54.88 | -43.09±54.88 | -268.95±114.45 | -43.18±54.96 | -43.18±54.96 | -43.18±54.96 | -43.18±54.96 |
| Task $f(x^*_{OFF})$ | | | | | | | GTOPX 6 -83.37±0.23 | | | | | | | | |
| ARCOO | -84.39±0.51 | 0.98±0.00 | 0.98±0.00 | 0.98±0.00 | 0.98±0.00 | -85.91±3.79 | 0.97±0.01 | 0.97±0.01 | 0.97±0.01 | 0.97±0.01 | -95.00±21.93 | 0.94±0.09 | 0.94±0.09 | 0.94±0.09 | 0.94±0.09 |
| BO | -179.81±20.08 | -0.66±0.17 | -0.66±0.17 | -0.66±0.17 | -0.66±0.17 | -165.50±15.39 | -0.69±0.15 | -0.69±0.15 | -0.69±0.15 | -0.69±0.15 | -182.71±33.61 | -0.68±0.14 | -0.68±0.14 | -0.68±0.14 | -0.68±0.14 |
| CBAS | -83.37±0.23 | 1.00±0.00 | 1.00±0.00 | 1.00±0.00 | 1.00±0.00 | -83.37±0.23 | 1.00±0.00 | 1.00±0.00 | 1.00±0.00 | 1.00±0.00 | -83.37±0.23 | 1.00±0.00 | 1.00±0.00 | 1.00±0.00 | 1.00±0.00 |
| CCDDEA | -182.12±167.98 | -1.98±2.90 | -2.99±4.36 | -2.37±3.48 | -2.55±3.74 | -7.11e3±1.66e4 | -57.22±102.07 | -98.93±186.12 | -72.27±131.84 | -66.56±120.30 | -7.12e3±1.66e4 | -67.79±111.90 | -110.04±180.63 | -83.58±137.38 | -67.96±111.78 |
| CMAES | -152.34±6.71 | -0.23±0.09 | -0.23±0.09 | -0.23±0.09 | -0.23±0.09 | - | - | - | - | - | - | - | - | - | - |
| TTDDEA | -1.12e3±2.26e3 | -17.20±35.07 | -17.20±35.07 | -17.20±35.07 | -17.20±35.07 | -1.62e3±2.54e3 | -22.27±43.24 | -22.26±43.25 | -22.26±43.25 | -22.27±43.25 | -1.08e3±1.64e3 | -32.17±64.74 | -32.17±64.74 | -32.17±64.74 | -32.17±64.74 |
| Trimentoring | -114.05±53.63 | -24.38±31.58 | -24.38±31.58 | -24.38±31.58 | -24.38±31.58 | -113.63±32.09 | -24.53±31.74 | -24.53±31.74 | -24.53±31.74 | -24.53±31.74 | -114.12±32.71 | -24.57±31.79 | -24.57±31.79 | -24.57±31.79 | -24.57±31.79 |

Table 25: Overall results for GTOPX unconstrained tasks with 128 solutions and 0th percentile evaluations. In this case, 0% of the values are missing near the worst value and another 50% near the optimal value. Details are the same as Table 5.

| Steps | | t = 50 | | | | | t = 100 | | | | | t = 150 | | | |
|---|---|---|---|---|---|---|---|---|---|---|---|---|---|---|---|
| Task $f(x^*_{OFF})$ | | | | | | | GTOPX 2 -356.70±2.33 | | | | | | | | |
| Metric | FS | SI | OI | SO | SO$_\omega$ | FS | SI | OI | SO | SO$_\omega$ | FS | SI | OI | SO | SO$_\omega$ |
| ARCOO | -394.31±85.68 | 0.81±0.11 | 0.81±0.11 | 0.81±0.11 | 0.81±0.11 | -362.49±2.34 | 0.88±0.06 | 0.88±0.06 | 0.88±0.06 | 0.88±0.06 | -362.49±2.34 | 0.91±0.04 | 0.91±0.04 | 0.91±0.04 | 0.91±0.04 |
| BO | -1.25e4±1.46e4 | -37.26±10.44 | -37.26±10.44 | -37.26±10.44 | -37.26±10.44 | -9.62e3±9.82e3 | -37.32±4.80 | -37.32±4.80 | -37.32±4.80 | -37.32±4.80 | -1.06e4±1.11e5 | -36.07±3.59 | -36.07±3.59 | -36.07±3.59 | -36.07±3.59 |
| CBAS | -356.70±2.33 | 1.00±0.00 | 1.00±0.00 | 1.00±0.00 | 1.00±0.00 | -356.70±2.33 | 1.00±0.00 | 1.00±0.00 | 1.00±0.00 | 1.00±0.00 | -356.70±2.33 | 1.00±0.00 | 1.00±0.00 | 1.00±0.00 | 1.00±0.00 |
| CCDDEA | -189.97±21.22 | 0.41±0.28 | 0.75±0.93 | 0.53±0.37 | 0.59±0.41 | -3.22e3±4.37e3 | -0.73±1.57 | -1.42±2.500 | -0.96±2.06 | -0.87±1.87 | -3.91e2±3.68e3 | -2.50±3.86 | -4.82±7.47 | -3.29±5.09 | -2.51±3.88 |
| CMAES | -359.17±71.32 | 0.55±0.18 | 0.68±0.21 | 0.61±0.09 | 0.63±0.20 | - | - | - | - | - | - | - | - | - | - |
| TTDDEA | -366.00±335.97 | -3.15±4.42 | -3.81±4.74 | -3.41±4.52 | -3.53±4.97 | -306.97±211.87 | -1.27±2.65 | -1.39±2.83 | -1.32±2.71 | -1.30±2.69 | -302.43±109.92 | -0.61±1.92 | -0.54±2.07 | -0.58±1.97 | -0.61±1.92 |
| Trimentoring | -307.33±42.54 | 0.85±0.06 | 1.00±0.15 | 0.92±0.09 | 0.94±0.10 | -307.11±42.57 | 0.89±0.05 | 1.06±0.15 | 0.97±0.09 | 0.94±0.07 | -384.10±171.06 | 0.90±0.04 | 1.06±0.15 | 0.97±0.08 | 0.90±0.04 |
| Task $f(x^*_{OFF})$ | | | | | | | GTOPX 3 -336.15±4.28 | | | | | | | | |
| ARCOO | -9.17e3±1.20e4 | -14.77±21.82 | -14.77±21.82 | -14.77±21.82 | -14.77±21.82 | -2.65e4±5.15e4 | -40.30±66.70 | -40.30±66.70 | -40.30±66.70 | -40.30±66.70 | -4.92e4±1.10e5 | -62.74±118.17 | -62.74±118.17 | -62.74±118.17 | -62.74±118.17 |
| BO | -5.68e4±3.35e4 | -614.81±432.41 | -614.81±432.41 | -614.81±432.41 | -614.81±432.41 | -5.65e4±5.79e4 | -474.50±209.86 | -474.50±209.86 | -474.50±209.86 | -474.50±209.86 | -3.31e4±1.42e4 | -975.29±1.31e3 | -975.29±1.31e3 | -975.29±1.31e3 | -975.29±1.31e3 |
| CBAS | -336.15±6.28 | 1.00±0.00 | 1.00±0.00 | 1.00±0.00 | 1.00±0.00 | -336.15±6.28 | 1.00±0.00 | 1.00±0.00 | 1.00±0.00 | 1.00±0.00 | -336.15±6.28 | 1.00±0.00 | 1.00±0.00 | 1.00±0.00 | 1.00±0.00 |
| CCDDEA | -2.47e4±6.60e4 | -64.91±101.34 | -117.56±187.51 | -83.38±131.51 | -92.37±146.23 | -1.03e3±1.62e3 | -35.74±49.99 | -65.84±91.77 | -46.27±64.24 | -42.23±58.62 | -510.86±400.24 | -23.67±32.86 | -43.59±60.05 | -30.65±42.96 | -23.75±32.96 |
| CMAES | -1.17e8±8.11e7 | -4.28e5±5.44e5 | -4.30e5±5.43e5 | -4.29e5±5.44e5 | -4.29e5±5.43e5 | - | - | - | - | - | - | - | - | - | - |
| TTDDEA | -5.72e6±1.02e7 | -4.91e3±7.57e3 | -4.91e3±7.57e3 | -4.91e3±7.57e3 | -4.91e3±7.57e3 | -2.62e6±6.11e6 | -2.20e4±5.35e4 | -2.20e4±5.35e4 | -2.20e4±5.35e4 | -2.20e4±5.35e4 | -9.50e7±1.68e8 | -4.89e4±7.24e4 | -4.89e4±7.28e4 | -4.89e4±7.28e4 | -4.89e4±7.28e4 |
| Trimentoring | -386.10±178.91 | -0.54±2.47 | -0.42±2.55 | -0.49±2.50 | -0.47±2.52 | -317.07±83.58 | 0.20±1.15 | 0.29±1.33 | 0.24±1.23 | 0.23±1.20 | -338.86±108.89 | 0.43±0.77 | 0.55±0.91 | 0.48±0.83 | 0.43±0.77 |
| Task $f(x^*_{OFF})$ | | | | | | | GTOPX 4 -496.56±5.01 | | | | | | | | |
| ARCOO | -1.37e3±1.79e3 | -1.24±4.42 | -1.24±4.42 | -1.24±4.42 | -1.24±4.42 | -1.45e3±1.83e3 | -1.24±4.41 | -1.24±4.41 | -1.24±4.41 | -1.24±4.41 | -1.50e3±1.97e3 | -1.33±4.51 | -1.33±4.51 | -1.33±4.51 | -1.33±4.51 |
| BO | -1.67e5±1.67e5 | -1.58e3±1.20e3 | -1.58e3±1.20e3 | -1.58e3±1.20e3 | -1.58e3±1.20e3 | -8.47e4±5.19e4 | -1.46e3±522.25 | -1.46e3±522.25 | -1.46e3±522.25 | -1.46e3±522.25 | -1.12e5±1.08e4 | -1.39e3±436.97 | -1.39e3±436.97 | -1.39e3±436.97 | -1.39e3±436.97 |
| CBAS | -496.56±5.01 | 1.00±0.00 | 1.00±0.00 | 1.00±0.00 | 1.00±0.00 | -496.56±5.01 | 1.00±0.00 | 1.00±0.00 | 1.00±0.00 | 1.00±0.00 | -496.56±5.01 | 1.00±0.00 | 1.00±0.00 | 1.00±0.00 | 1.00±0.00 |
| CCDDEA | -9.24e3±1.80e4 | -3.32e3±6.95e3 | -5.30e3±1.06e4 | -4.08e3±8.27e3 | -4.42e3±8.94e3 | -1.15e7±2.34e7 | -5.75e3±8.82e3 | -9.32e3±1.41e4 | -7.11e3±1.08e4 | -6.60e3±1.01e4 | -2.76e6±5.82e6 | -5.92e3±1.14e4 | -9.03e3±1.84e4 | -6.92e3±1.14e4 | -5.63e3±9.29e3 |
| CMAES | -1.49e9±1.23e9 | -1.43e6±1.17e6 | -1.43e6±1.17e6 | -1.43e6±1.17e6 | -1.43e6±1.17e6 | - | - | - | - | - | - | - | - | - | - |
| TTDDEA | -1.82e5±3.62e5 | -6.10e5±1.61e6 | -6.11e5±1.61e6 | -6.11e5±1.61e6 | -6.11e5±1.61e6 | -7.89e7±2.06e8 | -4.08e5±1.08e6 | -4.09e5±1.08e6 | -4.08e5±1.08e6 | -4.08e5±1.08e6 | -3.95e8±1.04e9 | -4.70e5±1.24e6 | -4.71e5±1.24e6 | -4.71e5±1.24e6 | -4.70e5±1.24e6 |
| Trimentoring | -1.53e4±3.87e4 | -4.27e3±1.11e4 | -4.27e3±1.11e4 | -4.27e3±1.11e4 | -4.27e3±1.11e4 | -5.32e3±1.24e4 | -2.12e3±5.53e3 | -2.12e3±5.53e3 | -2.12e3±5.53e3 | -2.12e3±5.53e3 | -5.48e3±1.24e4 | -1.41e3±3.66e3 | -1.41e3±3.66e3 | -1.41e3±3.66e3 | -1.41e3±3.66e3 |
| Task $f(x^*_{OFF})$ | | | | | | | GTOPX 6 -168.89±0.62 | | | | | | | | |
| ARCOO | -286.43±58.84 | -0.06±0.12 | -0.06±0.12 | -0.06±0.12 | -0.06±0.12 | -332.15±85.40 | -0.17±0.27 | -0.17±0.27 | -0.17±0.27 | -0.17±0.27 | -378.75±96.40 | -0.23±0.38 | -0.23±0.38 | -0.23±0.38 | -0.23±0.38 |
| BO | -5.71e2±3.88e3 | -42.77±5.58 | -42.77±5.58 | -42.77±5.58 | -42.77±5.58 | -3.67e3±1.42e3 | -45.51±7.43 | -45.51±7.43 | -45.51±7.43 | -45.51±7.43 | -4.15e3±2.16e4 | -44.66±5.81 | -44.66±5.81 | -44.66±5.81 | -44.66±5.81 |
| CBAS | -168.89±0.62 | 1.00±0.00 | 1.00±0.00 | 1.00±0.00 | 1.00±0.00 | -168.89±0.62 | 1.00±0.00 | 1.00±0.00 | 1.00±0.00 | 1.00±0.00 | -168.89±0.62 | 1.00±0.00 | 1.00±0.00 | 1.00±0.00 | 1.00±0.00 |
| CCDDEA | -212.95±123.28 | -0.05±0.71 | -0.00±1.15 | -0.04±0.88 | -0.03±0.95 | -579.63±480.00 | -0.56±1.17 | -0.82±1.84 | -0.67±1.45 | -0.63±1.23 | -579.63±480.00 | -0.87±1.57 | -1.33±2.49 | -1.06±1.93 | -0.88±1.58 |
| CMAES | -366.28±83.44 | -0.95±0.81 | -0.95±0.81 | -0.95±0.81 | -0.95±0.81 | - | - | - | - | - | - | - | - | - | - |
| TTDDEA | -1.43e3±1.76e3 | -12.76±16.54 | -13.93±15.99 | -13.23±16.32 | -13.43±16.22 | -1.89e3±2.99e3 | -11.35±15.52 | -11.83±15.23 | -11.54±15.40 | -11.47±15.44 | -1.28e4±2.85e4 | -20.06±35.10 | -20.36±34.95 | -20.18±39.08 | -20.06±35.10 |
| Trimentoring | -6.67e4±1.76e5 | -712.37±1.25e3 | -712.37±1.25e3 | -712.37±1.25e3 | -712.37±1.25e3 | -6.81e4±1.79e5 | -804.90±1.39e3 | -804.90±1.39e3 | -804.90±1.39e3 | -804.90±1.39e3 | -6.81e4±1.79e5 | -835.99±1.45e3 | -835.99±1.45e3 | -835.99±1.45e3 | -835.99±1.45e3 |

Table 26: Overall results for GTOPX unconstrained tasks with 128 solutions and 0th percentile evaluations. In this case, 0% of the values are missing near the worst value and another 40% near the optimal value. Details are the same as Table 5.

| Steps | | t = 50 | | | | | t = 100 | | | | | t = 150 | | | |
|---|---|---|---|---|---|---|---|---|---|---|---|---|---|---|---|
| Task $f(x^*_{OFF})$ | | | | | | | | GTOPX 2 $-275.27_{\pm2.44}$ | | | | | | | |
| Metric | FS | SI | OI | SO | SO$_\omega$ | FS | SI | OI | SO | SO$_\omega$ | FS | SI | OI | SO | SO$_\omega$ |
| ARCOO | $-282.53_{\pm7.53}$ | $0.75_{\pm0.07}$ | $0.75_{\pm0.07}$ | $0.75_{\pm0.07}$ | $0.75_{\pm0.07}$ | $-279.82_{\pm2.17}$ | $0.85_{\pm0.04}$ | $0.85_{\pm0.04}$ | $0.85_{\pm0.04}$ | $0.85_{\pm0.04}$ | $-305.18_{\pm65.97}$ | $0.88_{\pm0.03}$ | $0.88_{\pm0.03}$ | $0.88_{\pm0.03}$ | $0.88_{\pm0.03}$ |
| BO | $-1.16e4_{\pm9.46e3}$ | $-48.24_{\pm9.38}$ | $-48.24_{\pm9.38}$ | $-48.24_{\pm9.38}$ | $-48.24_{\pm9.38}$ | $-5.44e3_{\pm2.32e3}$ | $-50.35_{\pm6.33}$ | $-50.35_{\pm6.33}$ | $-50.35_{\pm6.33}$ | $-50.35_{\pm6.33}$ | $-1.00e4_{\pm5.44e3}$ | $-52.41_{\pm4.64}$ | $-52.41_{\pm4.64}$ | $-52.41_{\pm4.64}$ | $-52.41_{\pm4.64}$ |
| CBAS | $-275.27_{\pm2.34}$ | $1.00_{\pm0.00}$ | $1.00_{\pm0.00}$ | $1.00_{\pm0.00}$ | $1.00_{\pm0.00}$ | $-275.27_{\pm2.34}$ | $1.00_{\pm0.00}$ | $1.00_{\pm0.00}$ | $1.00_{\pm0.00}$ | $1.00_{\pm0.00}$ | $-275.27_{\pm2.34}$ | $1.00_{\pm0.00}$ | $1.00_{\pm0.00}$ | $1.00_{\pm0.00}$ | $1.00_{\pm0.00}$ |
| CCDDEA | $-307.86_{\pm168.32}$ | $0.34_{\pm0.55}$ | $0.60_{\pm0.61}$ | $0.43_{\pm0.45}$ | $0.48_{\pm0.49}$ | $-1.51e3_{\pm2.08e3}$ | $-1.98_{\pm5.26}$ | $-3.52_{\pm9.35}$ | $-2.53_{\pm6.74}$ | $-2.32_{\pm6.17}$ | $-3.93e3_{\pm5.94e3}$ | $-4.17_{\pm10.50}$ | $-7.39_{\pm18.67}$ | $-5.33_{\pm13.44}$ | $-4.18_{\pm10.53}$ |
| CMAES | $-350.36_{\pm38.75}$ | $0.23_{\pm0.41}$ | $0.23_{\pm0.44}$ | $0.23_{\pm0.42}$ | $0.23_{\pm0.43}$ | - | - | - | - | - | - | - | - | - | - |
| TTDDEA | $-1.63e3_{\pm3.45e3}$ | $-5.00_{\pm7.04}$ | $-5.37_{\pm6.87}$ | $-5.15_{\pm6.97}$ | $-5.21_{\pm6.94}$ | $-2.43e3_{\pm5.36e3}$ | $-5.66_{\pm2.11}$ | $-5.70_{\pm12.09}$ | $-5.67_{\pm12.10}$ | $-5.67_{\pm12.10}$ | $-1.22e3_{\pm3.93e3}$ | $-6.61_{\pm15.09}$ | $-6.54_{\pm15.12}$ | $-6.58_{\pm15.11}$ | $-6.61_{\pm15.09}$ |
| Trimentoring | $-866.85_{\pm990.68}$ | $0.09_{\pm1.29}$ | $0.09_{\pm1.33}$ | $0.09_{\pm1.31}$ | $0.09_{\pm1.31}$ | $-569.40_{\pm747.23}$ | $-0.14_{\pm2.11}$ | $-0.12_{\pm2.12}$ | $-0.13_{\pm2.12}$ | $-0.13_{\pm2.12}$ | $-567.43_{\pm748.07}$ | $-0.23_{\pm2.45}$ | $-0.20_{\pm2.46}$ | $-0.21_{\pm2.46}$ | $-0.23_{\pm2.45}$ |
| Task $f(x^*_{OFF})$ | | | | | | | | GTOPX 3 $-228.84_{\pm3.46}$ | | | | | | | |
| ARCOO | $-2.23e3_{\pm2.35e3}$ | $-16.91_{\pm28.93}$ | $-16.91_{\pm28.93}$ | $-16.91_{\pm28.93}$ | $-16.91_{\pm28.93}$ | $-3.17e3_{\pm4.42e3}$ | $-16.06_{\pm19.91}$ | $-16.06_{\pm19.91}$ | $-16.06_{\pm19.91}$ | $-16.06_{\pm19.91}$ | $-2.69e3_{\pm3.93e3}$ | $-15.55_{\pm19.63}$ | $-15.55_{\pm19.63}$ | $-15.55_{\pm19.63}$ | $-15.55_{\pm19.63}$ |
| BO | $-5.89e4_{\pm4.84e4}$ | $-718.67_{\pm508.53}$ | $-718.67_{\pm508.53}$ | $-718.67_{\pm508.53}$ | $-718.67_{\pm508.53}$ | $-6.77e4_{\pm8.99e4}$ | $-608.21_{\pm247.44}$ | $-608.21_{\pm247.44}$ | $-608.21_{\pm247.44}$ | $-608.21_{\pm247.44}$ | $-1.01e5_{\pm1.17e5}$ | $-604.51_{\pm194.09}$ | $-604.51_{\pm194.09}$ | $-604.51_{\pm194.09}$ | $-604.51_{\pm194.09}$ |
| CBAS | $-228.84_{\pm3.46}$ | $1.00_{\pm0.00}$ | $1.00_{\pm0.00}$ | $1.00_{\pm0.00}$ | $1.00_{\pm0.00}$ | $-228.84_{\pm3.46}$ | $1.00_{\pm0.00}$ | $1.00_{\pm0.00}$ | $1.00_{\pm0.00}$ | $1.00_{\pm0.00}$ | $-228.84_{\pm3.46}$ | $1.00_{\pm0.00}$ | $1.00_{\pm0.00}$ | $1.00_{\pm0.00}$ | $1.00_{\pm0.00}$ |
| CCDDEA | $-3.78e3_{\pm4.97e3}$ | $-5.25_{\pm6.30}$ | $-7.66_{\pm8.93}$ | $-6.15_{\pm7.29}$ | $-6.57_{\pm7.74}$ | $-294.21_{\pm339.28}$ | $-8.36_{\pm12.49}$ | $-14.59_{\pm23.68}$ | $-10.56_{\pm16.34}$ | $-9.72_{\pm14.85}$ | $-294.16_{\pm339.29}$ | $-5.40_{\pm8.61}$ | $-9.47_{\pm16.32}$ | $-6.84_{\pm11.26}$ | $-5.42_{\pm8.63}$ |
| CMAES | $-9.67e7_{\pm1.14e8}$ | $-1.14e5_{\pm1.52e5}$ | $-1.14e5_{\pm1.52e5}$ | $-1.14e5_{\pm1.52e5}$ | $-1.14e5_{\pm1.52e5}$ | - | - | - | - | - | - | - | - | - | - |
| TTDDEA | $-2.41e7_{\pm4.42e7}$ | $-5.23e4_{\pm8.52e4}$ | $-5.23e4_{\pm8.52e4}$ | $-5.23e4_{\pm8.52e4}$ | $-5.23e4_{\pm8.52e4}$ | $-1.40e7_{\pm3.40e7}$ | $-7.88e4_{\pm1.17e5}$ | $-7.88e4_{\pm1.17e5}$ | $-7.88e4_{\pm1.17e5}$ | $-7.88e4_{\pm1.17e5}$ | $-6.95e7_{\pm1.66e8}$ | $-1.80e5_{\pm2.22e5}$ | $-1.80e5_{\pm2.22e5}$ | $-1.80e5_{\pm2.22e5}$ | $-1.80e5_{\pm2.22e5}$ |
| Trimentoring | $-2.52e4_{\pm6.59e4}$ | $-428.27_{\pm822.55}$ | $-428.27_{\pm822.55}$ | $-428.27_{\pm822.55}$ | $-428.27_{\pm822.55}$ | $-3.21e4_{\pm8.43e4}$ | $-295.10_{\pm615.85}$ | $-295.10_{\pm615.85}$ | $-295.10_{\pm615.85}$ | $-295.10_{\pm615.85}$ | $-3.23e4_{\pm8.44e4}$ | $-254.04_{\pm559.17}$ | $-254.04_{\pm559.17}$ | $-254.04_{\pm559.17}$ | $-254.04_{\pm559.17}$ |
| Task $f(x^*_{OFF})$ | | | | | | | | GTOPX 4 $-322.50_{\pm1.08}$ | | | | | | | |
| ARCOO | $-1.34e3_{\pm1.83e1}$ | $-3.44_{\pm7.80}$ | $-3.44_{\pm7.80}$ | $-3.44_{\pm7.80}$ | $-3.44_{\pm7.80}$ | $-1.42e3_{\pm1.99e3}$ | $-3.78_{\pm8.53}$ | $-3.78_{\pm8.53}$ | $-3.78_{\pm8.53}$ | $-3.78_{\pm8.53}$ | $-1.43e3_{\pm2.01e3}$ | $-3.83_{\pm8.78}$ | $-3.83_{\pm8.78}$ | $-3.83_{\pm8.78}$ | $-3.83_{\pm8.78}$ |
| BO | $-5.17e5_{\pm9.95e5}$ | $-3.67e3_{\pm2.47e3}$ | $-3.67e3_{\pm2.47e3}$ | $-3.67e3_{\pm2.47e3}$ | $-3.67e3_{\pm2.47e3}$ | $-1.91e5_{\pm1.55e5}$ | $-3.28e3_{\pm1.84e3}$ | $-3.28e3_{\pm1.84e3}$ | $-3.28e3_{\pm1.84e3}$ | $-3.28e3_{\pm1.84e3}$ | $-2.23e5_{\pm2.11e5}$ | $-2.84e3_{\pm1.20e3}$ | $-2.84e3_{\pm1.20e3}$ | $-2.84e3_{\pm1.20e3}$ | $-2.84e3_{\pm1.20e3}$ |
| CBAS | $-322.50_{\pm1.08}$ | $1.00_{\pm0.00}$ | $1.00_{\pm0.00}$ | $1.00_{\pm0.00}$ | $1.00_{\pm0.00}$ | $-322.50_{\pm1.08}$ | $1.00_{\pm0.00}$ | $1.00_{\pm0.00}$ | $1.00_{\pm0.00}$ | $1.00_{\pm0.00}$ | $-322.50_{\pm1.08}$ | $1.00_{\pm0.00}$ | $1.00_{\pm0.00}$ | $1.00_{\pm0.00}$ | $1.00_{\pm0.00}$ |
| CCDDEA | $-1.16e5_{\pm2.18e5}$ | $-880.78_{\pm1.40e3}$ | $-1.01e3_{\pm1.45e3}$ | $-934.84_{\pm1.42e3}$ | $-957.34_{\pm1.41e3}$ | $-9.93e6_{\pm2.58e7}$ | $-4.06e4_{\pm7.22e4}$ | $-5.98e4_{\pm1.14e5}$ | $-4.81e4_{\pm8.83e4}$ | $-4.53e4_{\pm8.23e4}$ | $-3.41e5_{\pm4.06e5}$ | $-3.19e4_{\pm4.91e4}$ | $-4.54e4_{\pm7.59e4}$ | $-3.72e4_{\pm5.93e4}$ | $-3.19e4_{\pm4.92e4}$ |
| CMAES | $-9.22e8_{\pm1.66e9}$ | $-9.37e5_{\pm1.25e6}$ | $-9.37e5_{\pm1.25e6}$ | $-9.37e5_{\pm1.25e6}$ | $-9.37e5_{\pm1.25e6}$ | - | - | - | - | - | - | - | - | - | - |
| TTDDEA | $-5.13e4_{\pm1.27e5}$ | $-6.70e3_{\pm1.51e4}$ | $-6.77e3_{\pm1.51e4}$ | $-6.73e3_{\pm1.51e4}$ | $-6.74e3_{\pm1.51e4}$ | $-2.86e7_{\pm7.56e7}$ | $-1.36e4_{\pm3.46e4}$ | $-1.36e4_{\pm3.45e4}$ | $-1.36e4_{\pm3.46e4}$ | $-1.36e4_{\pm3.46e4}$ | $-1.93e5_{\pm4.01e5}$ | $-5.94e4_{\pm1.56e5}$ | $-5.94e4_{\pm1.56e5}$ | $-5.94e4_{\pm1.56e5}$ | $-5.94e4_{\pm1.56e5}$ |
| Trimentoring | $-1.96e3_{\pm2.28e3}$ | $-13.36_{\pm19.96}$ | $-13.36_{\pm19.96}$ | $-13.36_{\pm19.96}$ | $-13.36_{\pm19.96}$ | $-1.78e4_{\pm4.31e4}$ | $-14.34_{\pm20.08}$ | $-14.34_{\pm20.08}$ | $-14.34_{\pm20.08}$ | $-14.34_{\pm20.08}$ | $-1.38e3_{\pm1.88e3}$ | $-36.15_{\pm64.32}$ | $-36.15_{\pm64.32}$ | $-36.15_{\pm64.32}$ | $-36.15_{\pm64.32}$ |
| Task $f(x^*_{OFF})$ | | | | | | | | GTOPX 6 $-142.09_{\pm0.39}$ | | | | | | | |
| ARCOO | $-323.52_{\pm82.42}$ | $-0.87_{\pm0.73}$ | $-0.87_{\pm0.73}$ | $-0.87_{\pm0.73}$ | $-0.87_{\pm0.73}$ | $-324.91_{\pm80.47}$ | $-0.91_{\pm0.71}$ | $-0.91_{\pm0.71}$ | $-0.91_{\pm0.71}$ | $-0.91_{\pm0.71}$ | $-331.62_{\pm77.64}$ | $-0.93_{\pm0.71}$ | $-0.93_{\pm0.71}$ | $-0.93_{\pm0.71}$ | $-0.93_{\pm0.71}$ |
| BO | $-5.54e3_{\pm3.67e3}$ | $-54.72_{\pm10.26}$ | $-54.72_{\pm10.26}$ | $-54.72_{\pm10.26}$ | $-54.72_{\pm10.26}$ | $-6.94e3_{\pm4.35e3}$ | $-128.44_{\pm191.61}$ | $-128.44_{\pm191.61}$ | $-128.44_{\pm191.61}$ | $-128.44_{\pm191.61}$ | $-6.84e3_{\pm3.93e3}$ | $-106.49_{\pm128.02}$ | $-106.49_{\pm128.02}$ | $-106.49_{\pm128.02}$ | $-106.49_{\pm128.02}$ |
| CBAS | $-142.09_{\pm0.39}$ | $1.00_{\pm0.00}$ | $1.00_{\pm0.00}$ | $1.00_{\pm0.00}$ | $1.00_{\pm0.00}$ | $-142.09_{\pm0.39}$ | $1.00_{\pm0.00}$ | $1.00_{\pm0.00}$ | $1.00_{\pm0.00}$ | $1.00_{\pm0.00}$ | $-142.09_{\pm0.39}$ | $1.00_{\pm0.00}$ | $1.00_{\pm0.00}$ | $1.00_{\pm0.00}$ | $1.00_{\pm0.00}$ |
| CCDDEA | $-254.60_{\pm165.87}$ | $-3.42_{\pm7.85}$ | $-5.48_{\pm12.46}$ | $-4.21_{\pm9.63}$ | $-4.57_{\pm10.43}$ | $-266.05_{\pm188.71}$ | $-2.00_{\pm3.96}$ | $-3.17_{\pm6.29}$ | $-2.45_{\pm4.86}$ | $-2.29_{\pm4.53}$ | $-223.15_{\pm116.42}$ | $-1.49_{\pm2.74}$ | $-2.36_{\pm4.37}$ | $-1.82_{\pm3.37}$ | $-1.49_{\pm2.75}$ |
| CMAES | $-591.73_{\pm314.95}$ | $-1.37_{\pm0.62}$ | $-1.37_{\pm0.62}$ | $-1.37_{\pm0.62}$ | $-1.37_{\pm0.62}$ | - | - | - | - | - | - | - | - | - | - |
| TTDDEA | $-170.73_{\pm88.76}$ | $-3.31_{\pm3.12}$ | $-3.95_{\pm3.81}$ | $-3.58_{\pm3.40}$ | $-3.69_{\pm3.52}$ | $-192.01_{\pm68.02}$ | $-1.32_{\pm1.67}$ | $-1.53_{\pm2.09}$ | $-1.42_{\pm1.85}$ | $-1.38_{\pm1.78}$ | $-202.47_{\pm125.31}$ | $-0.76_{\pm1.21}$ | $-0.85_{\pm1.54}$ | $-0.80_{\pm1.35}$ | $-0.76_{\pm1.21}$ |
| Trimentoring | $-2.25e4_{\pm5.88e4}$ | $-3.75e3_{\pm8.67e3}$ | $-3.75e3_{\pm8.67e3}$ | $-3.75e3_{\pm8.67e3}$ | $-3.75e3_{\pm8.67e3}$ | $-2.27e4_{\pm5.93e4}$ | $-3.65e3_{\pm8.73e3}$ | $-3.65e3_{\pm8.73e3}$ | $-3.65e3_{\pm8.73e3}$ | $-3.65e3_{\pm8.73e3}$ | $-2.27e4_{\pm5.93e4}$ | $-3.62e3_{\pm8.75e3}$ | $-3.62e3_{\pm8.75e3}$ | $-3.62e3_{\pm8.75e3}$ | $-3.62e3_{\pm8.75e3}$ |

Table 27: Overall results for GTOPX unconstrained tasks with 128 solutions and 0th percentile evaluations. In this case, 0% of the values are missing near the worst value and another 30% near the optimal value. Details are the same as Table 5.

| Steps | | t = 50 | | | | | t = 100 | | | | | t = 150 | | | |
|---|---|---|---|---|---|---|---|---|---|---|---|---|---|---|---|
| Task $f(x^*_{OFF})$ | | | | | | | | GTOPX 2 $-218.12_{\pm1.79}$ | | | | | | | |
| Metric | FS | SI | OI | SO | SO$_\omega$ | FS | SI | OI | SO | SO$_\omega$ | FS | SI | OI | SO | SO$_\omega$ |
| ARCOO | $-280.96_{\pm15.44}$ | $0.37_{\pm0.30}$ | $0.37_{\pm0.30}$ | $0.37_{\pm0.30}$ | $0.37_{\pm0.30}$ | $-322.09_{\pm110.28}$ | $0.41_{\pm0.22}$ | $0.41_{\pm0.22}$ | $0.41_{\pm0.22}$ | $0.41_{\pm0.22}$ | $-279.21_{\pm17.62}$ | $0.46_{\pm0.16}$ | $0.46_{\pm0.16}$ | $0.46_{\pm0.16}$ | $0.46_{\pm0.16}$ |
| BO | $-9.36e3_{\pm5.33e3}$ | $-79.36_{\pm33.97}$ | $-79.36_{\pm33.97}$ | $-79.36_{\pm33.97}$ | $-79.36_{\pm33.97}$ | $-1.12e4_{\pm7.08e3}$ | $-77.94_{\pm18.45}$ | $-77.94_{\pm18.45}$ | $-77.94_{\pm18.45}$ | $-77.94_{\pm18.45}$ | $-1.14e4_{\pm4.61e3}$ | $-75.23_{\pm13.58}$ | $-75.23_{\pm13.58}$ | $-75.23_{\pm13.58}$ | $-75.23_{\pm13.58}$ |
| CBAS | $-218.12_{\pm1.78}$ | $1.00_{\pm0.00}$ | $1.00_{\pm0.00}$ | $1.00_{\pm0.00}$ | $1.00_{\pm0.00}$ | $-218.12_{\pm1.78}$ | $1.00_{\pm0.00}$ | $1.00_{\pm0.00}$ | $1.00_{\pm0.00}$ | $1.00_{\pm0.00}$ | $-218.12_{\pm1.78}$ | $1.00_{\pm0.00}$ | $1.00_{\pm0.00}$ | $1.00_{\pm0.00}$ | $1.00_{\pm0.00}$ |
| CCDDEA | $-320.27_{\pm274.70}$ | $-0.20_{\pm0.63}$ | $-0.36_{\pm1.19}$ | $-0.25_{\pm0.82}$ | $-0.28_{\pm0.92}$ | $-4.31e3_{\pm5.35e3}$ | $-5.35_{\pm5.46}$ | $-9.42_{\pm9.42}$ | $-6.80_{\pm6.86}$ | $-6.25_{\pm6.31}$ | $-7.55e3_{\pm1.36e4}$ | $-12.27_{\pm19.19}$ | $-22.91_{\pm37.36}$ | $-15.96_{\pm25.36}$ | $-12.30_{\pm19.25}$ |
| CMAES | $-369.16_{\pm32.65}$ | $-0.51_{\pm0.63}$ | $-0.51_{\pm0.63}$ | $-0.51_{\pm0.63}$ | $-0.51_{\pm0.63}$ | - | - | - | - | - | - | - | - | - | - |
| TTDDEA | $-279.43_{\pm122.57}$ | $-5.47_{\pm4.15}$ | $-5.71_{\pm4.08}$ | $-5.58_{\pm4.11}$ | $-5.62_{\pm4.10}$ | $-282.25_{\pm117.42}$ | $-2.45_{\pm2.41}$ | $-2.54_{\pm2.39}$ | $-2.49_{\pm2.40}$ | $-2.48_{\pm2.41}$ | $-264.75_{\pm86.27}$ | $-1.47_{\pm1.84}$ | $-1.50_{\pm1.84}$ | $-1.49_{\pm1.84}$ | $-1.47_{\pm1.84}$ |
| Trimentoring | $-358.75_{\pm208.85}$ | $-30.70_{\pm81.69}$ | $-30.70_{\pm81.69}$ | $-30.70_{\pm81.69}$ | $-30.70_{\pm81.69}$ | $-269.96_{\pm25.05}$ | $-32.31_{\pm81.67}$ | $-32.31_{\pm81.67}$ | $-32.31_{\pm81.67}$ | $-32.31_{\pm81.67}$ | $-275.87_{\pm29.30}$ | $-32.03_{\pm81.91}$ | $-32.03_{\pm81.91}$ | $-32.03_{\pm81.91}$ | $-32.03_{\pm81.91}$ |
| Task $f(x^*_{OFF})$ | | | | | | | | GTOPX 3 $-171.54_{\pm0.81}$ | | | | | | | |
| ARCOO | $-284.35_{\pm28.87}$ | $0.12_{\pm0.17}$ | $0.12_{\pm0.17}$ | $0.12_{\pm0.17}$ | $0.12_{\pm0.17}$ | $-597.65_{\pm917.70}$ | $-0.19_{\pm1.00}$ | $-0.19_{\pm1.00}$ | $-0.19_{\pm1.00}$ | $-0.19_{\pm1.00}$ | $-536.94_{\pm761.01}$ | $-0.88_{\pm2.96}$ | $-0.88_{\pm2.96}$ | $-0.88_{\pm2.96}$ | $-0.88_{\pm2.96}$ |
| BO | $-2.44e5_{\pm4.84e5}$ | $-818.17_{\pm371.55}$ | $-818.17_{\pm371.55}$ | $-818.17_{\pm371.55}$ | $-818.17_{\pm371.55}$ | $-2.71e5_{\pm5.15e5}$ | $-1.08e3_{\pm580.88}$ | $-1.08e3_{\pm580.88}$ | $-1.08e3_{\pm580.88}$ | $-1.08e3_{\pm580.88}$ | $-2.19e5_{\pm3.74e5}$ | $-1.03e3_{\pm340.24}$ | $-1.03e3_{\pm340.24}$ | $-1.03e3_{\pm340.24}$ | $-1.03e3_{\pm340.24}$ |
| CBAS | $-171.54_{\pm0.81}$ | $1.00_{\pm0.00}$ | $1.00_{\pm0.00}$ | $1.00_{\pm0.00}$ | $1.00_{\pm0.00}$ | $-171.54_{\pm0.81}$ | $1.00_{\pm0.00}$ | $1.00_{\pm0.00}$ | $1.00_{\pm0.00}$ | $1.00_{\pm0.00}$ | $-171.54_{\pm0.81}$ | $1.00_{\pm0.00}$ | $1.00_{\pm0.00}$ | $1.00_{\pm0.00}$ | $1.00_{\pm0.00}$ |
| CCDDEA | $-4.06e3_{\pm5.09e3}$ | $-33.42_{\pm39.06}$ | $-48.52_{\pm56.47}$ | $-38.94_{\pm44.73}$ | $-41.55_{\pm47.72}$ | $-692.56_{\pm1.07e3}$ | $-24.13_{\pm21.86}$ | $-37.58_{\pm34.83}$ | $-28.98_{\pm26.11}$ | $-27.13_{\pm24.61}$ | $-734.97_{\pm1.09e3}$ | $-16.81_{\pm14.08}$ | $-26.11_{\pm22.72}$ | $-20.17_{\pm16.89}$ | $-16.84_{\pm14.11}$ |
| CMAES | $-5.73e7_{\pm6.97e7}$ | $-1.83e5_{\pm2.79e5}$ | $-1.83e5_{\pm2.79e5}$ | $-1.83e5_{\pm2.79e5}$ | $-1.83e5_{\pm2.79e5}$ | - | - | - | - | - | - | - | - | - | - |
| TTDDEA | $-7.27e7_{\pm1.88e8}$ | $-6.78e4_{\pm1.53e5}$ | $-6.78e4_{\pm1.53e5}$ | $-6.78e4_{\pm1.53e5}$ | $-6.78e4_{\pm1.53e5}$ | $-2.20e8_{\pm4.63e8}$ | $-2.82e5_{\pm6.15e5}$ | $-2.82e5_{\pm6.15e5}$ | $-2.82e5_{\pm6.15e5}$ | $-2.82e5_{\pm6.15e5}$ | $-1.95e8_{\pm3.33e8}$ | $-8.80e5_{\pm1.64e6}$ | $-8.80e5_{\pm1.64e6}$ | $-8.80e5_{\pm1.64e6}$ | $-8.80e5_{\pm1.64e6}$ |
| Trimentoring | $-8.14e4_{\pm1.43e5}$ | $-2.51e5_{\pm6.64e5}$ | $-2.51e5_{\pm6.64e5}$ | $-2.51e5_{\pm6.64e5}$ | $-2.51e5_{\pm6.64e5}$ | $-4.95e5_{\pm1.31e6}$ | $-1.35e5_{\pm3.26e5}$ | $-1.35e5_{\pm3.26e5}$ | $-1.35e5_{\pm3.26e5}$ | $-1.35e5_{\pm3.26e5}$ | $-3.45e3_{\pm3.36e3}$ | $-8.97e4_{\pm2.16e5}$ | $-8.97e4_{\pm2.16e5}$ | $-8.97e4_{\pm2.16e5}$ | $-8.97e4_{\pm2.16e5}$ |
| Task $f(x^*_{OFF})$ | | | | | | | | GTOPX 4 $-241.61_{\pm1.39}$ | | | | | | | |
| ARCOO | $-1.73e3_{\pm1.97e3}$ | $-7.04_{\pm11.03}$ | $-7.04_{\pm11.03}$ | $-7.04_{\pm11.03}$ | $-7.04_{\pm11.03}$ | $-1.71e3_{\pm1.95e3}$ | $-7.68_{\pm11.36}$ | $-7.68_{\pm11.36}$ | $-7.68_{\pm11.36}$ | $-7.68_{\pm11.36}$ | $-1.74e3_{\pm1.91e3}$ | $-7.89_{\pm11.49}$ | $-7.89_{\pm11.49}$ | $-7.89_{\pm11.49}$ | $-7.89_{\pm11.49}$ |
| BO | $-1.87e5_{\pm1.99e5}$ | $-2.50e3_{\pm1.32e3}$ | $-2.50e3_{\pm1.32e3}$ | $-2.50e3_{\pm1.32e3}$ | $-2.50e3_{\pm1.32e3}$ | $-1.40e6_{\pm3.41e6}$ | $-3.25e3_{\pm891.92}$ | $-3.25e3_{\pm891.92}$ | $-3.25e3_{\pm891.92}$ | $-3.25e3_{\pm891.92}$ | $-1.70e5_{\pm1.91e4}$ | $-5.16e3_{\pm3.04e3}$ | $-5.16e3_{\pm3.04e3}$ | $-5.16e3_{\pm3.04e3}$ | $-5.16e3_{\pm3.04e3}$ |
| CBAS | $-241.61_{\pm1.39}$ | $1.00_{\pm0.00}$ | $1.00_{\pm0.00}$ | $1.00_{\pm0.00}$ | $1.00_{\pm0.00}$ | $-241.61_{\pm1.39}$ | $1.00_{\pm0.00}$ | $1.00_{\pm0.00}$ | $1.00_{\pm0.00}$ | $1.00_{\pm0.00}$ | $-241.61_{\pm1.39}$ | $1.00_{\pm0.00}$ | $1.00_{\pm0.00}$ | $1.00_{\pm0.00}$ | $1.00_{\pm0.00}$ |
| CCDDEA | $-2.05e4_{\pm4.54e4}$ | $-1.05e5_{\pm2.71e5}$ | $-1.05e5_{\pm2.71e5}$ | $-1.05e5_{\pm2.71e5}$ | $-1.05e5_{\pm2.71e5}$ | $-2.75e5_{\pm2.91e5}$ | $-4.26e5_{\pm9.76e5}$ | $-5.73e5_{\pm1.36e6}$ | $-4.87e5_{\pm1.14e6}$ | $-4.76e5_{\pm1.14e6}$ | $-5.79e5_{\pm9.88e5}$ | $-3.52e5_{\pm8.16e5}$ | $-4.76e5_{\pm1.14e6}$ | $-4.04e5_{\pm9.51e5}$ | $-3.52e5_{\pm8.18e5}$ |
| CMAES | $-1.62e9_{\pm2.05e9}$ | $-8.90e5_{\pm1.89e6}$ | $-8.90e5_{\pm1.89e6}$ | $-8.90e5_{\pm1.89e6}$ | $-8.90e5_{\pm1.89e6}$ | $-5.55e3_{\pm1.34e4}$ | $-856.24_{\pm1.01e3}$ | $-914.83_{\pm1.02e3}$ | $-880.67_{\pm1.02e3}$ | $-871.90_{\pm1.01e3}$ | $-5.21e3_{\pm1.25e4}$ | $-669.47_{\pm889.39}$ | $-708.87_{\pm890.38}$ | $-685.82_{\pm888.86}$ | $-669.66_{\pm889.38}$ |
| TTDDEA | $-4.36e3_{\pm1.01e4}$ | $-1.74e3_{\pm1.97e3}$ | $-1.81e3_{\pm2.09e3}$ | $-1.77e3_{\pm1.98e3}$ | $-1.78e3_{\pm1.99e3}$ | $-6.89e8_{\pm1.82e9}$ | $-3.17e6_{\pm7.96e6}$ | $-3.17e6_{\pm7.96e6}$ | $-3.17e6_{\pm7.96e6}$ | $-3.17e6_{\pm7.96e6}$ | $-1.47e10_{\pm3.89e10}$ | $-1.52e7_{\pm3.59e7}$ | $-1.52e7_{\pm3.59e7}$ | $-1.52e7_{\pm3.59e7}$ | $-1.52e7_{\pm3.59e7}$ |
| Trimentoring | $-1.78e8_{\pm4.72e8}$ | $-1.98e6_{\pm4.42e6}$ | $-1.98e6_{\pm4.42e6}$ | $-1.98e6_{\pm4.42e6}$ | $-1.98e6_{\pm4.42e6}$ | $-6.89e8_{\pm1.82e9}$ | $-3.17e6_{\pm7.96e6}$ | $-3.17e6_{\pm7.96e6}$ | $-3.17e6_{\pm7.96e6}$ | $-3.17e6_{\pm7.96e6}$ | $-1.47e10_{\pm3.89e10}$ | $-1.52e7_{\pm3.59e7}$ | $-1.52e7_{\pm3.59e7}$ | $-1.52e7_{\pm3.59e7}$ | $-1.52e7_{\pm3.59e7}$ |
| Task $f(x^*_{OFF})$ | | | | | | | | GTOPX 6 $-121.48_{\pm0.38}$ | | | | | | | |
| ARCOO | $-486.07_{\pm478.11}$ | $-1.22_{\pm0.54}$ | $-1.22_{\pm0.54}$ | $-1.22_{\pm0.54}$ | $-1.22_{\pm0.54}$ | $-288.70_{\pm20.71}$ | $-1.32_{\pm0.52}$ | $-1.32_{\pm0.52}$ | $-1.32_{\pm0.52}$ | $-1.32_{\pm0.52}$ | $-293.17_{\pm31.18}$ | $-1.38_{\pm0.57}$ | $-1.38_{\pm0.57}$ | $-1.38_{\pm0.57}$ | $-1.38_{\pm0.57}$ |
| BO | $-4.98e3_{\pm5.78e3}$ | $-72.04_{\pm10.30}$ | $-72.04_{\pm10.30}$ | $-72.04_{\pm10.30}$ | $-72.04_{\pm10.30}$ | $-5.37e3_{\pm5.06e3}$ | $-76.28_{\pm7.41}$ | $-76.28_{\pm7.41}$ | $-76.28_{\pm7.41}$ | $-76.28_{\pm7.41}$ | $-4.70e3_{\pm3.09e3}$ | $-74.75_{\pm7.27}$ | $-74.75_{\pm7.27}$ | $-74.75_{\pm7.27}$ | $-74.75_{\pm7.27}$ |
| CBAS | $-121.48_{\pm0.38}$ | $1.00_{\pm0.00}$ | $1.00_{\pm0.00}$ | $1.00_{\pm0.00}$ | $1.00_{\pm0.00}$ | $-121.48_{\pm0.38}$ | $1.00_{\pm0.00}$ | $1.00_{\pm0.00}$ | $1.00_{\pm0.00}$ | $1.00_{\pm0.00}$ | $-121.48_{\pm0.38}$ | $1.00_{\pm0.00}$ | $1.00_{\pm0.00}$ | $1.00_{\pm0.00}$ | $1.00_{\pm0.00}$ |
| CCDDEA | $-158.65_{\pm27.70}$ | $-0.77_{\pm1.07}$ | $-1.03_{\pm1.41}$ | $-0.88_{\pm1.22}$ | $-0.92_{\pm1.28}$ | $-211.91_{\pm117.45}$ | $-0.37_{\pm0.57}$ | $-0.50_{\pm0.86}$ | $-0.43_{\pm0.68}$ | $-0.41_{\pm0.64}$ | $-208.17_{\pm119.56}$ | $-0.25_{\pm0.43}$ | $-0.30_{\pm0.63}$ | $-0.26_{\pm0.51}$ | - |
| CMAES | $-439.23_{\pm182.50}$ | $-2.23_{\pm0.21}$ | $-2.23_{\pm0.21}$ | $-2.23_{\pm0.21}$ | $-2.23_{\pm0.21}$ | $-1.79e3_{\pm3.88e3}$ | $-12.22_{\pm14.37}$ | $-14.72_{\pm17.95}$ | $-13.19_{\pm15.55}$ | $-12.83_{\pm15.08}$ | $-2.81e3_{\pm3.43e3}$ | $-17.95_{\pm26.61}$ | $-19.58_{\pm26.95}$ | $-18.58_{\pm26.67}$ | $-17.96_{\pm26.61}$ |
| TTDDEA | $-360.55_{\pm440.10}$ | $-24.61_{\pm33.88}$ | $-25.02_{\pm33.64}$ | $-24.80_{\pm33.77}$ | $-24.87_{\pm33.73}$ | $-289.84_{\pm84.95}$ | $-1.45_{\pm0.66}$ | $-1.45_{\pm0.66}$ | $-1.45_{\pm0.66}$ | $-1.45_{\pm0.66}$ | $-1.04e4_{\pm2.67e4}$ | $-2.51_{\pm2.89}$ | $-2.51_{\pm2.89}$ | $-2.51_{\pm2.89}$ | $-2.51_{\pm2.89}$ |
| Trimentoring | $-300.86_{\pm79.24}$ | $-1.49_{\pm0.73}$ | $-1.49_{\pm0.73}$ | $-1.49_{\pm0.73}$ | $-1.49_{\pm0.73}$ | $-289.84_{\pm84.95}$ | $-1.45_{\pm0.66}$ | $-1.45_{\pm0.66}$ | $-1.45_{\pm0.66}$ | $-1.45_{\pm0.66}$ | $-1.04e4_{\pm2.67e4}$ | $-2.51_{\pm2.89}$ | $-2.51_{\pm2.89}$ | $-2.51_{\pm2.89}$ | $-2.51_{\pm2.89}$ |

Table 28: Overall results for GTOPX unconstrained tasks with 128 solutions and 0th percentile evaluations. In this case, 0% of the values are missing near the worst value and another 20% near the optimal value. Details are the same as Table 5.

| Steps | t = 50 | | | | | t = 100 | | | | | t = 150 | | | | |
|---|---|---|---|---|---|---|---|---|---|---|---|---|---|---|---|
| **Task** $f(x^*_{OFF})$ | GTOPX 2 −175.10±0.75 | | | | | | | | | | | | | | |
| Metric | FS | SI | OI | SO | SO_w | FS | SI | OI | SO | SO_w | FS | SI | OI | SO | SO_w |
| ARCOO | −263.85±26.39 | 0.06±0.14 | 0.06±0.14 | 0.06±0.14 | 0.06±0.14 | −261.73±25.12 | 0.08±0.21 | 0.08±0.21 | 0.08±0.21 | 0.08±0.21 | −256.50±25.56 | 0.06±0.26 | 0.06±0.26 | 0.06±0.26 | 0.06±0.26 |
| BO | −9.29e3±6.51e3 | −90.93±10.53 | −90.93±10.53 | −90.93±10.53 | −90.93±10.53 | −8.39e3±5.78e3 | −93.03±9.66 | −93.03±9.66 | −93.03±9.66 | −93.03±9.66 | −9.29e3±7.00e3 | −92.54±8.81 | −92.54±8.81 | −92.54±8.81 | −92.54±8.81 |
| CBAS | −175.10±0.75 | 1.00±0.00 | 1.00±0.00 | 1.00±0.00 | 1.00±0.00 | −175.10±0.75 | 1.00±0.00 | 1.00±0.00 | 1.00±0.00 | 1.00±0.00 | −175.10±0.75 | 1.00±0.00 | 1.00±0.00 | 1.00±0.00 | 1.00±0.00 |
| CCDDEA | −460.03±362.02 | −0.36±0.59 | −0.60±1.00 | −0.45±0.73 | −0.49±0.81 | −2.00e3±3.01e3 | −3.00±4.50 | −5.21±9.00 | −3.80±5.74 | −3.50±5.27 | −2.04e3±3.08e3 | −5.15±8.37 | −8.81±14.31 | −6.48±10.51 | −5.16±8.39 |
| CMAES | −343.95±39.45 | −0.62±0.29 | −0.62±0.29 | −0.62±0.29 | −0.62±0.29 | - | - | - | - | - | - | - | - | - | - |
| TTDDEA | −1.91e3±4.28e3 | −32.94±49.21 | −32.94±49.21 | −32.94±49.21 | −32.94±49.21 | −6.29e3±1.60e4 | −34.84±72.63 | −34.86±72.62 | −34.85±72.62 | −34.84±72.62 | −8.28e3±2.10e4 | −41.68±96.93 | −41.69±96.92 | −41.68±96.92 | −41.68±96.93 |
| Trimentoring | −8.93e3±1.92e4 | −38.04±86.03 | −38.04±86.03 | −38.04±86.03 | −38.04±86.03 | −9.18e3±1.9e4 | −55.60±122.99 | −55.60±122.99 | −55.60±122.99 | −55.60±122.99 | −9.19e3±1.9e4 | −61.53±135.31 | −61.53±135.31 | −61.53±135.31 | −61.53±135.31 |
| **Task** $f(x^*_{OFF})$ | GTOPX 3 −134.83±0.56 | | | | | | | | | | | | | | |
| ARCOO | −288.19±83.08 | −0.53±0.35 | −0.53±0.35 | −0.53±0.35 | −0.53±0.35 | −2.47e3±5.82e3 | −9.59±24.34 | −9.59±24.34 | −9.59±24.34 | −9.59±24.34 | −2.80e3±6.72e3 | −14.17±36.42 | −14.17±36.42 | −14.17±36.42 | −14.17±36.42 |
| BO | −7.03e4±3.45e4 | −1.57e3±881.38 | −1.57e3±881.38 | −1.57e3±881.38 | −1.57e3±881.38 | −4.50e4±4.16e4 | −1.63e3±643.79 | −1.63e3±643.79 | −1.63e3±643.79 | −1.63e3±643.79 | −7.48e4±7.00e4 | −1.44e3±473.66 | −1.44e3±473.66 | −1.44e3±473.66 | −1.44e3±473.66 |
| CBAS | −134.83±0.56 | 1.00±0.00 | 1.00±0.00 | 1.00±0.00 | 1.00±0.00 | −134.83±0.56 | 1.00±0.00 | 1.00±0.00 | 1.00±0.00 | 1.00±0.00 | −134.83±0.56 | 1.00±0.00 | 1.00±0.00 | 1.00±0.00 | 1.00±0.00 |
| CCDDEA | −4.75e5±1.29e6 | −2.11e4±5.55e4 | −2.11e4±5.55e4 | −2.11e4±5.55e4 | −2.11e4±5.55e4 | −665.73±1.11e3 | −1.07e4±2.75e4 | −1.09e4±2.75e4 | −1.08e4±2.75e4 | −1.08e4±2.75e4 | −631.18±1.12e3 | −6.53e3±1.68e4 | −7.24e3±1.83e4 | −6.86e3±1.78e4 | −6.54e3±1.68e4 |
| CMAES | −1.20e8±1.42e8 | −3.18e5±5.47e5 | −3.18e5±5.47e5 | −3.18e5±5.47e5 | −3.18e5±5.47e5 | - | - | - | - | - | - | - | - | - | - |
| TTDDEA | −1.32e8±2.21e8 | −5.76e5±1.22e6 | −5.76e5±1.22e6 | −5.76e5±1.22e6 | −5.76e5±1.22e6 | −8.96e7±3.87e7 | −1.04e6±1.72e6 | −1.04e6±1.72e6 | −1.04e6±1.72e6 | −1.04e6±1.72e6 | −2.19e8±4.20e8 | −6.08e6±1.38e7 | −6.08e6±1.38e7 | −6.08e6±1.38e7 | −6.08e6±1.38e7 |
| Trimentoring | −2.88e3±6.12e3 | −7.17e3±1.22e4 | −7.17e3±1.22e4 | −7.17e3±1.22e4 | −7.17e3±1.22e4 | −4.78e5±1.26e6 | −3.49e4±8.51e4 | −3.49e4±8.51e4 | −3.49e4±8.51e4 | −3.49e4±8.51e4 | −612.30±761.97 | −3.11e4±7.74e4 | −3.11e4±7.74e4 | −3.11e4±7.74e4 | −3.11e4±7.74e4 |
| **Task** $f(x^*_{OFF})$ | GTOPX 4 −194.03±0.88 | | | | | | | | | | | | | | |
| ARCOO | −2.72e3±2.69e3 | −18.61±19.38 | −18.61±19.38 | −18.61±19.38 | −18.61±19.38 | −2.75e3±2.73e3 | −19.66±20.94 | −19.66±20.94 | −19.66±20.94 | −19.66±20.94 | −2.74e3±2.73e3 | −20.07±21.48 | −20.07±21.48 | −20.07±21.48 | −20.07±21.48 |
| BO | −2.70e5±3.49e6 | −4.40e3±2.36e3 | −4.40e3±2.36e3 | −4.40e3±2.36e3 | −4.40e3±2.36e3 | −4.26e5±9.46e5 | −4.23e3±740.62 | −4.23e3±740.62 | −4.23e3±740.62 | −4.23e3±740.62 | −1.89e5±1.87e5 | −3.79e4±8.75e4 | −3.79e4±8.75e4 | −3.79e4±8.75e4 | −3.79e4±8.75e4 |
| CBAS | −194.03±0.88 | 1.00±0.00 | 1.00±0.00 | 1.00±0.00 | 1.00±0.00 | −194.03±0.88 | 1.00±0.00 | 1.00±0.00 | 1.00±0.00 | 1.00±0.00 | −194.03±0.88 | 1.00±0.00 | 1.00±0.00 | 1.00±0.00 | 1.00±0.00 |
| CCDDEA | −1.47e6±3.88e6 | −8.94e5±2.36e6 | −8.95e5±2.36e6 | −8.94e5±2.36e6 | −8.94e5±2.36e6 | −4.37e7±1.66e7 | −5.73e5±1.15e6 | −5.92e5±1.36e6 | −5.82e5±1.15e6 | −5.79e5±1.15e6 | −3.00e6±4.26e6 | −3.93e5±7.63e5 | −4.06e5±7.67e5 | −3.99e5±7.65e5 | −3.93e5±7.63e5 |
| CMAES | −1.50e8±1.63e8 | −1.49e6±3.21e6 | −1.49e6±3.21e6 | −1.49e6±3.21e6 | −1.49e6±3.21e6 | - | - | - | - | - | - | - | - | - | - |
| TTDDEA | −8.62e4±1.72e5 | −1.69e5±4.25e5 | −1.69e5±4.25e5 | −1.69e5±4.25e5 | −1.69e5±4.25e5 | −2.68e5±6.62e5 | −1.05e5±2.68e5 | −1.05e5±2.68e5 | −1.05e5±2.68e5 | −1.05e5±2.68e5 | −1.66e5±2.93e5 | −7.44e4±1.89e5 | −7.44e4±1.89e5 | −7.44e4±1.89e5 | −7.44e4±1.89e5 |
| Trimentoring | −1.26e3±1.44e3 | −156.97±391.01 | −156.97±391.01 | −156.97±391.01 | −156.97±391.01 | −1.33e3±1.52e3 | −81.82±192.75 | −81.82±192.75 | −81.82±192.75 | −81.82±192.75 | −1.30e3±1.61e3 | −57.24±127.72 | −57.24±127.72 | −57.24±127.72 | −57.24±127.72 |
| **Task** $f(x^*_{OFF})$ | GTOPX 6 −102.96±0.29 | | | | | | | | | | | | | | |
| ARCOO | −282.58±17.92 | −1.66±0.40 | −1.66±0.40 | −1.66±0.40 | −1.66±0.40 | −289.34±17.05 | −1.98±0.30 | −1.98±0.30 | −1.98±0.30 | −1.98±0.30 | −297.22±14.88 | −1.99±0.23 | −1.99±0.23 | −1.99±0.23 | −1.99±0.23 |
| BO | −5.44e3±3.52e3 | −87.29±19.47 | −87.29±19.47 | −87.29±19.47 | −87.29±19.47 | −5.61e3±4.17e3 | −86.72±11.10 | −86.72±11.10 | −86.72±11.10 | −86.72±11.10 | −7.80e3±6.26e3 | −88.20±3.61 | −88.20±3.61 | −88.20±3.61 | −88.20±3.61 |
| CBAS | −102.96±0.29 | 1.00±0.00 | 1.00±0.00 | 1.00±0.00 | 1.00±0.00 | −102.96±0.29 | 1.00±0.00 | 1.00±0.00 | 1.00±0.00 | 1.00±0.00 | −102.96±0.29 | 1.00±0.00 | 1.00±0.00 | 1.00±0.00 | 1.00±0.00 |
| CCDDEA | −272.75±205.66 | −0.60±0.69 | −0.78±0.87 | −0.67±0.75 | −0.70±0.79 | −6.79e3±1.75e4 | −17.23±43.64 | −33.20±85.37 | −22.65±57.77 | −20.54±52.28 | −6.79e3±1.75e4 | −32.17±83.50 | −62.44±163.20 | −42.43±110.49 | −32.27±63.78 |
| CMAES | −486.46±211.96 | −3.27±0.86 | −3.27±0.86 | −3.27±0.86 | −3.27±0.86 | - | - | - | - | - | - | - | - | - | - |
| TTDDEA | −193.68±180.33 | −7.22±6.58 | −7.81±6.58 | −7.46±6.58 | −7.57±6.56 | −184.53±82.42 | −3.68±3.85 | −3.98±3.88 | −3.81±3.85 | −3.76±3.85 | −199.75±77.85 | −2.51±2.67 | −2.73±2.73 | −2.60±2.68 | −2.51±2.67 |
| Trimentoring | −260.51±118.64 | −544.06±1.36e3 | −544.06±1.36e3 | −544.06±1.36e3 | −544.06±1.36e3 | −238.34±94.95 | −546.75±1.36e3 | −546.75±1.36e3 | −546.75±1.36e3 | −546.75±1.36e3 | −241.61±98.16 | −547.64±1.37e3 | −547.64±1.37e3 | −547.64±1.37e3 | −547.64±1.37e3 |

Table 29: Overall results for GTOPX unconstrained tasks with 128 solutions and 0th percentile evaluations. In this case, 0% of the values are missing near the worst value and another 10% near the optimal value. Details are the same as Table 5.

| Steps | t = 50 | | | | | t = 100 | | | | | t = 150 | | | | |
|---|---|---|---|---|---|---|---|---|---|---|---|---|---|---|---|
| **Task** $f(x^*_{OFF})$ | GTOPX 2 −136.28±0.39 | | | | | | | | | | | | | | |
| Metric | FS | SI | OI | SO | SO_w | FS | SI | OI | SO | SO_w | FS | SI | OI | SO | SO_w |
| ARCOO | −302.76±109.44 | −0.80±0.33 | −0.80±0.33 | −0.80±0.33 | −0.80±0.33 | −264.05±21.74 | −0.79±0.32 | −0.79±0.32 | −0.79±0.32 | −0.79±0.32 | −273.49±30.11 | −0.83±0.32 | −0.83±0.32 | −0.83±0.32 | −0.83±0.32 |
| BO | −5.93e3±2.15e4 | −127.30±10.39 | −127.30±10.39 | −127.30±10.39 | −127.30±10.39 | −1.09e4±1.06e5 | −128.78±14.02 | −128.78±14.02 | −128.78±14.02 | −128.78±14.02 | −1.62e4±2.33e4 | −131.03±12.51 | −131.03±12.51 | −131.03±12.51 | −131.03±12.51 |
| CBAS | −136.28±0.39 | 1.00±0.00 | 1.00±0.00 | 1.00±0.00 | 1.00±0.00 | −136.28±0.39 | 1.00±0.00 | 1.00±0.00 | 1.00±0.00 | 1.00±0.00 | −136.28±0.39 | 1.00±0.00 | 1.00±0.00 | 1.00±0.00 | 1.00±0.00 |
| CCDDEA | −345.68±187.72 | −1.11±2.57 | −1.14±2.57 | −1.12±2.57 | −1.13±2.57 | −564.93±515.81 | −1.61±1.69 | −2.40±2.06 | −1.88±1.77 | −1.78±1.73 | −553.40±522.57 | −1.87±1.80 | −3.10±3.22 | −2.29±2.24 | −1.87±1.80 |
| CMAES | −432.95±140.51 | −1.56±0.62 | −1.56±0.62 | −1.56±0.62 | −1.56±0.62 | - | - | - | - | - | - | - | - | - | - |
| TTDDEA | −436.01±337.98 | −18.49±8.01 | −18.49±8.01 | −18.49±8.01 | −18.49±8.01 | −678.05±797.40 | −12.04±7.87 | −12.04±7.87 | −12.04±7.87 | −12.04±7.87 | −707.71±823.83 | −11.44±11.22 | −11.44±11.22 | −11.44±11.22 | −11.44±11.22 |
| Trimentoring | −259.67±41.49 | −2.92±4.40 | −2.92±4.40 | −2.92±4.40 | −2.92±4.40 | −299.35±114.32 | −1.77±2.31 | −1.77±2.31 | −1.77±2.31 | −1.77±2.31 | −309.44±104.32 | −1.40±1.66 | −1.40±1.66 | −1.40±1.66 | −1.40±1.66 |
| **Task** $f(x^*_{OFF})$ | GTOPX 3 −102.69±0.49 | | | | | | | | | | | | | | |
| ARCOO | −286.14±27.07 | −1.69±0.42 | −1.69±0.42 | −1.69±0.42 | −1.69±0.42 | −277.29±31.50 | −1.81±0.45 | −1.81±0.45 | −1.81±0.45 | −1.81±0.45 | −278.14±37.22 | −1.78±0.43 | −1.78±0.43 | −1.78±0.43 | −1.78±0.43 |
| BO | −3.18e4±2.78e4 | −2.14e3±1.13e3 | −2.14e3±1.13e3 | −2.14e3±1.13e3 | −2.14e3±1.13e3 | −7.87e4±6.92e4 | −1.74e3±577.42 | −1.74e3±577.42 | −1.74e3±577.42 | −1.74e3±577.42 | −2.82e4±1.76e4 | −1.96e3±750.31 | −1.96e3±750.31 | −1.96e3±750.31 | −1.96e3±750.31 |
| CBAS | −102.69±0.49 | 1.00±0.00 | 1.00±0.00 | 1.00±0.00 | 1.00±0.00 | −102.69±0.49 | 1.00±0.00 | 1.00±0.00 | 1.00±0.00 | 1.00±0.00 | −102.69±0.49 | 1.00±0.00 | 1.00±0.00 | 1.00±0.00 | 1.00±0.00 |
| CCDDEA | −8.30e4±2.13e5 | −1.64e3±3.40e3 | −1.64e3±3.40e3 | −1.64e3±3.40e3 | −1.64e3±3.40e3 | −184.06±264.43 | −554.43±1.31e3 | −842.54±2.03e3 | −667.26±1.39e3 | −625.64±1.48e3 | −171.15±71.70 | −368.03±507.71 | −559.95±1.35e3 | −443.22±1.08e3 | −368.85±869.76 |
| CMAES | −5.13e7±8.57e7 | −2.86e5±4.96e5 | −2.86e5±4.96e5 | −2.86e5±4.96e5 | −2.86e5±4.96e5 | - | - | - | - | - | - | - | - | - | - |
| TTDDEA | −9.02e7±1.25e8 | −5.80e5±7.06e5 | −5.80e5±7.06e5 | −5.80e5±7.06e5 | −5.80e5±7.06e5 | −1.85e8±5.08e8 | −2.36e6±2.55e6 | −2.36e6±2.55e6 | −2.36e6±2.55e6 | −2.36e6±2.55e6 | −5.89e8±9.24e8 | −8.98e6±1.24e7 | −8.98e6±1.24e7 | −8.98e6±1.24e7 | −8.98e6±1.24e7 |
| Trimentoring | −655.05±739.36 | −9.14e3±2.42e4 | −9.14e3±2.42e4 | −9.14e3±2.42e4 | −9.14e3±2.42e4 | −4.50e7±1.19e8 | −1.81e4±4.71e4 | −1.81e4±4.71e4 | −1.81e4±4.71e4 | −1.81e4±4.71e4 | −4.71e4±1.23e5 | −1.11e5±2.93e5 | −1.11e5±2.93e5 | −1.11e5±2.93e5 | −1.11e5±2.93e5 |
| **Task** $f(x^*_{OFF})$ | GTOPX 4 −153.62±0.60 | | | | | | | | | | | | | | |
| ARCOO | −793.64±171.73 | −6.73±1.40 | −6.73±1.40 | −6.73±1.40 | −6.73±1.40 | −919.72±391.73 | −6.87±1.41 | −6.87±1.41 | −6.87±1.41 | −6.87±1.41 | −892.43±275.60 | −6.98±1.39 | −6.98±1.39 | −6.98±1.39 | −6.98±1.39 |
| BO | −3.54e5±3.41e6 | −6.68e3±2.91e3 | −6.68e3±2.91e3 | −6.68e3±2.91e3 | −6.68e3±2.91e3 | −1.13e5±7.74e4 | −7.68e3±4.26e3 | −7.68e3±4.26e3 | −7.68e3±4.26e3 | −7.68e3±4.26e3 | −1.13e5±4.75e4 | −7.80e3±3.42e3 | −7.80e3±3.42e3 | −7.80e3±3.42e3 | −7.80e3±3.42e3 |
| CBAS | −153.62±0.60 | 1.00±0.00 | 1.00±0.00 | 1.00±0.00 | 1.00±0.00 | −153.62±0.60 | 1.00±0.00 | 1.00±0.00 | 1.00±0.00 | 1.00±0.00 | −153.62±0.60 | 1.00±0.00 | 1.00±0.00 | 1.00±0.00 | 1.00±0.00 |
| CCDDEA | −3.12e3±4.00e3 | −2.20e4±3.94e4 | −2.20e4±3.94e4 | −2.20e4±3.94e4 | −2.20e4±3.94e4 | −7.32e8±1.82e9 | −1.41e6±2.12e6 | −1.54e6±2.19e6 | −1.46e6±2.13e6 | −1.44e6±2.11e6 | −1.65e7±2.33e7 | −1.88e6±2.85e6 | −1.90e6±2.85e6 | −1.84e6±2.85e6 | −1.80e6±2.85e6 |
| CMAES | −2.81e8±1.83e8 | −1.21e6±1.80e5 | −1.21e6±1.80e5 | −1.21e6±1.80e5 | −1.21e6±1.80e5 | - | - | - | - | - | - | - | - | - | - |
| TTDDEA | −7.41e6±1.99e7 | −3.14e5±7.56e5 | −3.14e5±7.56e5 | −3.14e5±7.56e5 | −3.14e5±7.56e5 | −6.29e6±1.66e7 | −4.56e5±1.17e6 | −4.56e5±1.17e6 | −4.56e5±1.17e6 | −4.56e5±1.17e6 | −4.93e8±1.30e9 | −2.65e6±6.99e6 | −2.65e6±6.99e6 | −2.65e6±6.99e6 | −2.65e6±6.99e6 |
| Trimentoring | −714.53±75.42 | −6.33±1.37 | −6.33±1.37 | −6.33±1.37 | −6.33±1.37 | −773.21±104.49 | −6.01±0.58 | −6.01±0.58 | −6.01±0.58 | −6.01±0.58 | −764.00±37.59 | −6.14±0.52 | −6.14±0.52 | −6.14±0.52 | −6.14±0.52 |
| **Task** $f(x^*_{OFF})$ | GTOPX 6 −83.37±0.23 | | | | | | | | | | | | | | |
| ARCOO | −310.18±45.91 | −3.16±0.31 | −3.16±0.31 | −3.16±0.31 | −3.16±0.31 | −369.45±163.54 | −3.28±0.36 | −3.28±0.36 | −3.28±0.36 | −3.28±0.36 | −316.45±46.99 | −3.40±0.43 | −3.40±0.43 | −3.40±0.43 | −3.40±0.43 |
| BO | −4.76e3±3.75e3 | −124.53±19.92 | −124.53±19.92 | −124.53±19.92 | −124.53±19.92 | −7.11e3±6.48e3 | −129.19±22.10 | −129.19±22.10 | −129.19±22.10 | −129.19±22.10 | −1.02e4±7.78e3 | −127.16±15.52 | −127.16±15.52 | −127.16±15.52 | −127.16±15.52 |
| CBAS | −83.37±0.23 | 1.00±0.00 | 1.00±0.00 | 1.00±0.00 | 1.00±0.00 | −83.37±0.23 | 1.00±0.00 | 1.00±0.00 | 1.00±0.00 | 1.00±0.00 | −83.37±0.23 | 1.00±0.00 | 1.00±0.00 | 1.00±0.00 | 1.00±0.00 |
| CCDDEA | −228.78±180.36 | −2.17±1.28 | −2.92±2.41 | −2.46±1.70 | −2.59±1.89 | −157.74±59.14 | −1.73±0.87 | −2.35±1.49 | −1.99±1.20 | −1.56±0.93 | −155.24±60.74 | −1.82±1.20 | | | −1.39±0.80 |
| CMAES | −489.44±377.62 | −4.80±1.11 | −4.80±1.11 | −4.80±1.11 | −4.80±1.11 | - | - | - | - | - | - | - | - | - | - |
| TTDDEA | −184.97±107.56 | −13.16±7.89 | −13.70±7.13 | −13.38±7.46 | −13.47±7.36 | −221.78±180.71 | −7.58±5.02 | −7.84±4.78 | −7.68±4.91 | −7.64±4.95 | −910.12±2.12e3 | −6.82±5.51 | −7.01±5.36 | −6.89±4.45 | −6.82±5.51 |
| Trimentoring | −6.20e4±1.44e5 | −598.76±1.34e3 | −598.76±1.34e3 | −598.76±1.34e3 | −598.76±1.34e3 | −6.22e4±1.44e5 | −923.77±2.10e3 | −923.77±2.10e3 | −923.77±2.10e3 | −923.77±2.10e3 | −6.18e4±1.43e5 | −1.03e3±2.35e3 | −1.03e3±2.35e3 | −1.03e3±2.35e3 | −1.03e3±2.35e3 |

Table 30: Overall results for GTOPX unconstrained tasks with 128 solutions and 0th percentile evaluations. In this case, 10% of the values are missing near the worst value and another 40% near the optimal value. Details are the same as Table 5.

| Steps | | t = 50 | | | | | t = 100 | | | | | t = 150 | | | |
|---|---|---|---|---|---|---|---|---|---|---|---|---|---|---|---|
| Task $f(x^*_{OFF})$ | | | | | | GTOPX 2 −275.27±2.34 | | | | | | | | | |
| Metric | FS | SI | OI | SO | SO_ω | FS | SI | OI | SO | SO_ω | FS | SI | OI | SO | SO_ω |
| ARCOO | -1.12e3±1.64e3 | -1.56±3.38 | -1.56±3.38 | -1.56±3.38 | -1.56±3.38 | -1.12e3±1.55e3 | -2.35±5.60 | -2.35±5.60 | -2.35±5.60 | -2.35±5.80 | -1.07e3±1.21e3 | -2.49±5.77 | -2.49±5.77 | -2.49±5.77 | -2.49±5.77 |
| BO | -8.15e3±6.73e3 | -49.42±7.74 | -49.42±7.74 | -49.42±7.74 | -49.42±7.74 | -9.11e3±6.03e3 | -62.17±33.41 | -62.17±33.41 | -62.17±33.41 | -62.17±33.41 | -725.44±520.61 | -0.29±0.94 | -0.46±1.58 | -60.08±23.48 | -60.08±23.48 |
| CBAS | -275.27±2.34 | 1.00±0.00 | 1.00±0.00 | 1.00±0.00 | 1.00±0.00 | -275.27±2.34 | 1.00±0.00 | 1.00±0.00 | 1.00±0.00 | 1.00±0.00 | -275.27±2.34 | 1.00±0.00 | 1.00±0.00 | 1.00±0.00 | 1.00±0.00 |
| CCDDEA | -476.07±221.66 | 0.35±0.29 | 0.60±0.52 | 0.44±0.37 | 0.49±0.41 | -876.72±601.25 | -0.00±0.62 | 1.00±0.00 | 0.00±0.77 | -0.00±0.71 | -725.44±520.61 | -0.29±0.94 | -0.46±1.58 | -0.36±1.17 | -0.29±0.94 |
| CMAES | -436.19±121.82 | 0.14±0.54 | 0.15±0.35 | 0.15±0.35 | 0.15±0.35 | | | | | | | | | | |
| TTDDEA | -316.28±175.88 | -8.03±15.73 | -8.46±15.98 | -8.21±15.67 | -8.28±15.64 | -255.21±47.99 | -3.48±7.93 | -3.69±7.89 | -3.56±7.91 | -3.53±7.92 | -257.60±68.03 | -2.02±5.30 | -2.08±5.31 | -2.04±5.30 | -2.02±5.30 |
| Trimentoring | -277.05±11.86 | -43.13±116.23 | -43.10±116.24 | -43.12±116.23 | -43.11±116.23 | -277.37±11.16 | -43.40±116.82 | -43.37±116.83 | -43.38±116.83 | -43.39±116.82 | -276.77±10.63 | -43.43±117.03 | -43.40±117.05 | -43.42±117.04 | -43.43±117.03 |
| Task $f(x^*_{OFF})$ | | | | | | GTOPX 3 −228.84±3.46 | | | | | | | | | |
| ARCOO | -289.15±38.93 | 0.38±0.39 | 0.38±0.39 | 0.38±0.39 | 0.38±0.39 | -273.93±22.21 | 0.53±0.23 | 0.53±0.23 | 0.53±0.23 | 0.53±0.23 | -261.51±21.90 | 0.61±0.16 | 0.61±0.16 | 0.61±0.16 | 0.61±0.16 |
| BO | -7.42e4±7.09e4 | -1.28e3±1.13e3 | -1.28e3±1.13e3 | -1.28e3±1.13e3 | -1.28e3±1.13e3 | -3.72e5±6.51e5 | -919.42±541.12 | -919.42±541.12 | -919.42±541.12 | -919.42±541.12 | -818.96±341.81 | -818.96±341.81 | -818.96±341.81 | -818.96±341.81 | -818.96±341.81 |
| CBAS | -228.84±3.46 | 1.00±0.00 | 1.00±0.00 | 1.00±0.00 | 1.00±0.00 | -228.84±3.46 | 1.00±0.00 | 1.00±0.00 | 1.00±0.00 | 1.00±0.00 | -228.84±3.46 | 1.00±0.00 | 1.00±0.00 | 1.00±0.00 | 1.00±0.00 |
| CCDDEA | -180.77±97.84 | -2.42±3.48 | -4.25±5.15 | -3.08±4.45 | -3.40±4.91 | -357.55±478.35 | -1.15±1.79 | -2.08±3.22 | -1.48±2.50 | -1.35±2.11 | -352.69±459.37 | -0.73±1.49 | -1.30±2.65 | -0.93±1.91 | -0.73±1.49 |
| CMAES | -378.25±82.22 | -31.74±45.47 | -31.74±45.47 | -31.74±45.47 | -31.74±45.47 | | | | | | | | | | |
| TTDDEA | -2.55e3±3.30e3 | -95.89±91.04 | -102.47±91.47 | -98.57±91.14 | -99.71±91.23 | -2.37e5±4.80e5 | -427.57±930.38 | -432.77±908.50 | -429.76±909.50 | -428.98±909.81 | -5.03e8±1.30e9 | -3.59e4±7.39e4 | -3.59e4±7.39e4 | -3.59e4±7.39e4 | -3.59e4±7.39e4 |
| Trimentoring | -5.72e7±1.51e8 | -1.66e5±2.40e5 | -1.66e5±2.40e5 | -1.66e5±2.40e5 | -1.66e5±2.40e5 | -3.19e7±4.48e7 | -2.64e5±5.97e5 | -2.64e5±5.97e5 | -2.64e5±5.97e5 | -2.64e5±5.97e5 | -9.09e7±2.41e8 | -3.66e5±9.67e5 | -3.66e5±9.67e5 | -3.66e5±9.67e5 | -3.66e5±9.67e5 |
| Task $f(x^*_{OFF})$ | | | | | | GTOPX 4 −322.50±1.08 | | | | | | | | | |
| ARCOO | -1.46e3±1.54e3 | -2.04±3.09 | -2.04±3.09 | -2.04±3.09 | -2.04±3.09 | -1.93e3±2.32e3 | -3.51±5.57 | -3.51±5.57 | -3.51±5.57 | -3.51±5.57 | -1.93e3±2.32e3 | -4.16±6.73 | -4.16±6.73 | -4.16±6.73 | -4.16±6.73 |
| BO | -1.47e5±1.98e4 | -3.35e3±3.53e3 | -3.35e3±3.52e3 | -3.35e3±3.52e3 | -3.35e3±3.52e3 | -3.51e5±6.42e5 | -5.80e3±6.26e3 | -5.80e3±6.26e3 | -5.80e3±6.26e3 | -5.80e3±6.26e3 | -4.80e4±3.91e4 | -4.50e3±4.22e3 | -4.50e3±4.22e3 | -4.50e3±4.22e3 | -4.50e3±4.22e3 |
| CBAS | -322.50±1.08 | 1.00±0.00 | 1.00±0.00 | 1.00±0.00 | 1.00±0.00 | -322.50±1.08 | 1.00±0.00 | 1.00±0.00 | 1.00±0.00 | 1.00±0.00 | -322.50±1.08 | 1.00±0.00 | 1.00±0.00 | 1.00±0.00 | 1.00±0.00 |
| CCDDEA | -748.96±546.33 | -12.48±10.57 | -20.94±18.66 | -15.54±13.39 | -16.99±14.77 | -5.53e5±7.90e5 | -1.57e4±2.56e4 | -2.40e4±3.86e4 | -1.87e4±2.95e4 | -1.76e4±2.80e4 | -9.28e5±1.75e6 | -1.25e4±2.09e4 | -1.87e4±2.88e4 | -1.48e4±2.38e4 | -1.25e4±2.09e4 |
| CMAES | -1.12e3±710.60 | -37.52±33.14 | -37.52±33.14 | -37.52±33.14 | -37.52±33.14 | | | | | | | | | | |
| TTDDEA | -1.28e5±2.29e5 | -3.17e5±8.23e5 | -3.17e5±8.23e5 | -3.17e5±8.23e5 | -3.17e5±8.23e5 | -1.71e5±3.18e5 | -2.86e5±7.47e5 | -2.86e5±7.47e5 | -2.86e5±7.47e5 | -2.86e5±7.47e5 | -2.49e5±4.20e5 | -2.80e5±7.34e5 | -2.80e5±7.34e5 | -2.80e5±7.34e5 | -2.80e5±7.34e5 |
| Trimentoring | -782.04±159.69 | -1.18±1.00 | -1.18±1.00 | -1.18±1.00 | -1.18±1.00 | -750.77±89.31 | -0.97±0.66 | -0.97±0.66 | -0.97±0.66 | -0.97±0.66 | -723.12±98.19 | -0.94±0.55 | -0.94±0.55 | -0.94±0.55 | -0.94±0.55 |
| Task $f(x^*_{OFF})$ | | | | | | GTOPX 4 −142.09±0.39 | | | | | | | | | |
| ARCOO | -1.53e3±2.04e3 | -6.51±7.56 | -6.51±7.56 | -6.51±7.56 | -6.51±7.56 | -1.83e3±3.20e3 | -10.37±18.23 | -10.37±18.23 | -10.37±18.23 | -10.37±18.23 | -2.42e3±3.62e3 | -13.39±23.29 | -13.39±23.29 | -13.39±23.29 | -13.39±23.29 |
| BO | -7.18e3±6.01e5 | -56.54±6.22 | -56.54±6.22 | -56.54±6.22 | -56.54±6.22 | -4.57e3±4.64e3 | -58.56±13.88 | -58.56±13.88 | -58.56±13.88 | -58.56±13.88 | -1.14e4±1.54e4 | -60.62±19.19 | -60.62±19.19 | -60.62±19.19 | -60.62±19.19 |
| CBAS | -142.09±0.39 | 1.00±0.00 | 1.00±0.00 | 1.00±0.00 | 1.00±0.00 | -142.09±0.39 | 1.00±0.00 | 1.00±0.00 | 1.00±0.00 | 1.00±0.00 | -142.09±0.39 | 1.00±0.00 | 1.00±0.00 | 1.00±0.00 | 1.00±0.00 |
| CCDDEA | -364.76±381.76 | -0.94±1.18 | -1.19±1.29 | -1.04±1.21 | -1.08±1.23 | -790.62±717.34 | -2.08±2.17 | -3.29±3.63 | -2.52±2.71 | -2.36±2.51 | -612.38±810.60 | -1.42±2.52 | -2.68±4.74 | -1.86±3.28 | -1.42±2.53 |
| CMAES | -346.70±34.90 | -1.41±0.32 | -1.41±0.32 | -1.41±0.32 | -1.41±0.32 | | | | | | | | | | |
| TTDDEA | -127.61±41.78 | -3.14±2.69 | -4.36±3.78 | -3.64±3.15 | -3.86±3.33 | -129.50±30.70 | -1.15±1.20 | -1.66±1.80 | -1.36±1.44 | -1.28±1.35 | -152.71±32.02 | -0.52±0.77 | -0.75±1.16 | -0.61±0.93 | -0.52±0.77 |
| Trimentoring | -8.11e3±1.00e4 | -32.13±40.19 | -32.13±40.19 | -32.13±40.19 | -32.13±40.19 | -8.69e3±1.11e4 | -54.65±70.13 | -54.65±70.13 | -54.65±70.13 | -54.65±70.13 | -8.70e3±1.11e4 | -62.52±80.86 | -62.52±80.86 | -62.52±80.86 | -62.52±80.86 |

Table 31: Overall results for GTOPX unconstrained tasks with 128 solutions and 0th percentile evaluations. In this case, 20% of the values are missing near the worst value and another 30% near the optimal value. Details are the same as Table 5.

| Steps | | t = 50 | | | | | t = 100 | | | | | t = 150 | | | |
|---|---|---|---|---|---|---|---|---|---|---|---|---|---|---|---|
| Task $f(x^*_{OFF})$ | | | | | | GTOPX 2 −218.12±1.78 | | | | | | | | | |
| Metric | FS | SI | OI | SO | SO_ω | FS | SI | OI | SO | SO_ω | FS | SI | OI | SO | SO_ω |
| ARCOO | -620.28±164.64 | -1.07±0.75 | -1.07±0.75 | -1.07±0.75 | -1.07±0.75 | -610.64±263.90 | -1.18±0.98 | -1.18±0.98 | -1.18±0.98 | -1.18±0.98 | -963.53±787.27 | -1.63±1.53 | -1.63±1.53 | -1.63±1.53 | -1.63±1.53 |
| BO | -2.03e4±1.34e4 | -67.56±12.95 | -67.56±12.95 | -67.56±12.95 | -67.56±12.95 | -3.18e4±4.43e4 | -69.60±10.34 | -69.60±10.34 | -69.60±10.34 | -69.60±10.34 | -7.78e3±4.36e3 | -67.60±8.69 | -67.60±8.69 | -67.60±8.69 | -67.60±8.69 |
| CBAS | -218.12±1.78 | 1.00±0.00 | 1.00±0.00 | 1.00±0.00 | 1.00±0.00 | -218.12±1.78 | 1.00±0.00 | 1.00±0.00 | 1.00±0.00 | 1.00±0.00 | -218.12±1.78 | 1.00±0.00 | 1.00±0.00 | 1.00±0.00 | 1.00±0.00 |
| CCDDEA | -236.93±44.76 | 0.18±0.45 | 0.37±0.68 | 0.24±0.53 | 0.28±0.56 | -1.23e3±1.26e3 | -13.35±30.95 | -19.60±44.06 | -15.85±36.37 | -14.94±34.42 | -838.99±438.94 | -9.64±20.91 | -14.40±29.69 | -11.52±24.55 | -9.66±20.95 |
| CMAES | -381.39±63.41 | -0.23±0.44 | -0.23±0.44 | -0.23±0.44 | -0.23±0.44 | | | | | | | | | | |
| TTDDEA | -329.93±124.20 | -10.21±9.99 | -10.76±10.97 | -10.47±10.45 | -10.56±10.62 | -333.20±108.52 | -4.81±4.89 | -5.19±5.36 | -4.98±5.11 | -4.92±5.14 | -412.65±126.20 | -3.04±3.26 | -3.38±3.55 | -3.19±3.30 | -3.04±3.26 |
| Trimentoring | -274.74±38.98 | -73.12±126.97 | -73.12±126.97 | -73.12±126.97 | -73.12±126.97 | -268.69±29.15 | -73.41±127.70 | -73.41±127.70 | -73.41±127.70 | -73.41±127.70 | -1.22e3±2.40e3 | -73.77±127.78 | -73.77±127.78 | -73.77±127.78 | -73.77±127.78 |
| Task $f(x^*_{OFF})$ | | | | | | GTOPX 3 −171.54±0.81 | | | | | | | | | |
| ARCOO | -545.00±477.61 | -0.70±1.18 | -0.70±1.18 | -0.70±1.18 | -0.70±1.18 | -877.16±1.15e3 | -1.74±3.02 | -1.74±3.02 | -1.74±3.02 | -1.74±3.02 | -1.02e3±1.18e3 | -2.62±4.55 | -2.62±4.55 | -2.62±4.55 | -2.62±4.55 |
| BO | -3.24e4±2.19e4 | -1.06e3±976.82 | -1.06e3±976.82 | -1.06e3±976.82 | -1.06e3±976.82 | -9.52e4±1.15e5 | -846.34±576.81 | -846.34±576.81 | -846.34±576.81 | -846.34±576.81 | -1.20e5±1.07e5 | -802.62±358.12 | -802.62±358.12 | -802.62±358.12 | -802.62±358.12 |
| CBAS | -171.54±0.81 | 1.00±0.00 | 1.00±0.00 | 1.00±0.00 | 1.00±0.00 | -171.54±0.81 | 1.00±0.00 | 1.00±0.00 | 1.00±0.00 | 1.00±0.00 | -171.54±0.81 | 1.00±0.00 | 1.00±0.00 | 1.00±0.00 | 1.00±0.00 |
| CCDDEA | -301.15±292.51 | -1.84±2.21 | -3.14±4.92 | -2.32±2.58 | -2.54±2.72 | -508.82±545.69 | -1.56±2.91 | -2.85±5.23 | -2.01±3.73 | -1.84±3.42 | -612.38±810.60 | -1.42±2.52 | -2.68±4.74 | -1.86±3.28 | -1.43±2.53 |
| CMAES | -1.40e3±2.36e3 | -24.03±34.02 | -24.03±34.02 | -24.03±34.02 | -24.03±34.02 | | | | | | | | | | |
| TTDDEA | -1.55e4±3.78e4 | -1.26e3±3.16e3 | -1.27e3±3.16e3 | -1.26e3±3.16e3 | -1.26e3±3.16e3 | -1.49e5±2.05e5 | -1.21e3±2.00e3 | -1.22e3±2.00e3 | -1.22e3±2.00e3 | -1.21e3±2.00e3 | -3.95e7±3.38e7 | -1.34e4±2.62e4 | -1.35e4±2.62e4 | -1.34e4±2.62e4 | -1.34e4±2.62e4 |
| Trimentoring | -1.37e4±3.43e4 | -39.79±97.76 | -39.79±97.76 | -39.79±97.76 | -39.79±97.76 | -3.26e5±8.26e5 | -105.88±193.21 | -105.88±193.21 | -105.88±193.21 | -105.88±193.21 | -3.16e5±7.93e5 | -1.08e3±2.60e3 | -1.08e3±2.60e3 | -1.08e3±2.60e3 | -1.08e3±2.60e3 |
| Task $f(x^*_{OFF})$ | | | | | | GTOPX 4 −241.61±1.30 | | | | | | | | | |
| ARCOO | -1.82e3±1.17e3 | -4.26±2.84 | -4.26±2.84 | -4.26±2.84 | -4.26±2.84 | -2.61e3±1.83e3 | -7.33±5.66 | -7.33±5.66 | -7.33±5.66 | -7.33±5.66 | -3.43e3±2.61e3 | -10.35±8.09 | -10.35±8.09 | -10.35±8.09 | -10.35±8.09 |
| BO | -1.09e6±1.92e6 | -3.30e3±1.12e3 | -3.30e3±1.12e3 | -3.30e3±1.12e3 | -3.30e3±1.12e3 | -1.89e5±2.73e4 | -3.53e3±1.13e3 | -3.53e3±1.13e3 | -3.53e3±1.13e3 | -3.53e3±1.13e3 | -1.36e5±1.42e9 | -4.00e3±1.75e3 | -4.00e3±1.75e3 | -4.00e3±1.75e3 | -4.00e3±1.75e3 |
| CBAS | -241.61±1.39 | 1.00±0.00 | 1.00±0.00 | 1.00±0.00 | 1.00±0.00 | -241.61±1.39 | 1.00±0.00 | 1.00±0.00 | 1.00±0.00 | 1.00±0.00 | -241.61±1.39 | 1.00±0.00 | 1.00±0.00 | 1.00±0.00 | 1.00±0.00 |
| CCDDEA | -939.43±621.75 | -57.62±90.47 | -67.22±96.89 | -61.34±99.19 | -63.01±99.05 | -422.92±456.00 | -592.17±641.96 | -491.35±530.61 | -466.50±503.47 | -491.35±530.61 | -1.87e5±2.65e5 | -632.08±782.76 | -866.57±1.03e3 | -728.70±886.87 | -633.17±783.97 |
| CMAES | -711.77±55.61 | -25.43±32.40 | -25.43±32.40 | -25.43±32.40 | -25.43±32.40 | | | | | | | | | | |
| TTDDEA | -3.16e3±7.28e3 | -422.74±620.00 | -434.52±619.32 | -428.11±619.62 | -430.14±619.51 | -243.30±242.15 | -257.77±239.81 | -250.38±240.77 | -248.53±241.20 | -5.33e3±3.78e4 | -215.25±260.26 | -223.52±257.60 | -218.61±258.99 | -215.28±260.22 | |
| Trimentoring | -1.46e3±2.24e3 | -42.83±103.78 | -42.83±103.78 | -42.83±103.78 | -42.83±103.78 | -1.48e3±2.2e3 | -44.74±103.96 | -44.74±103.96 | -44.74±103.96 | -44.74±103.96 | -1.47e3±2.28e3 | -45.38±104.05 | -45.38±104.05 | -45.38±104.05 | -45.38±104.05 |
| Task $f(x^*_{OFF})$ | | | | | | GTOPX 6 −121.48±0.39 | | | | | | | | | |
| ARCOO | -2.16e3±2.72e3 | -10.79±10.00 | -10.79±10.00 | -10.79±10.00 | -10.79±10.00 | -1.55e4±3.5.6e4 | -54.89±114.69 | -54.89±114.69 | -54.89±114.69 | -54.89±114.69 | -1.13e5±2.08e5 | -279.66±694.69 | -279.66±694.69 | -279.66±694.69 | -279.66±694.69 |
| BO | -5.90e3±4.16e3 | -76.04±27.38 | -76.04±27.38 | -76.04±27.38 | -76.04±27.38 | -7.08e3±5.31e3 | -73.06±21.21 | -73.06±21.21 | -73.06±21.21 | -73.06±21.21 | -72.07±13.16 | -72.07±13.16 | -72.07±13.16 | -72.07±13.16 | -72.07±13.16 |
| CBAS | -121.48±0.38 | 1.00±0.00 | 1.00±0.00 | 1.00±0.00 | 1.00±0.00 | -121.48±0.38 | 1.00±0.00 | 1.00±0.00 | 1.00±0.00 | 1.00±0.00 | -121.48±0.38 | 1.00±0.00 | 1.00±0.00 | 1.00±0.00 | 1.00±0.00 |
| CCDDEA | -210.19±125.64 | -0.88±1.38 | -1.33±2.04 | -1.08±1.64 | -1.13±1.76 | -577.97±471.31 | -1.80±1.84 | -2.63±2.21 | -2.12±2.00 | -1.80±1.84 | -562.98±487.53 | -2.15±2.69 | -3.48±3.57 | -2.78±3.00 | -2.36±2.69 |
| CMAES | -404.30±119.01 | -5.31±8.66 | -5.31±8.66 | -5.31±8.66 | -5.31±8.66 | -141.24±92.19 | -1.22±1.57 | -1.45±1.72 | -1.32±1.64 | -1.28±1.61 | -138.35±77.13 | -0.54±1.05 | -0.70±1.29 | -0.60±1.14 | -0.54±1.05 |
| TTDDEA | -139.96±35.65 | -3.20±3.15 | -3.76±3.20 | -3.43±3.16 | -3.55±3.17 | | | | | | | | | | |
| Trimentoring | -1.54e5±2.99e5 | -1.05e3±1.79e3 | -1.05e3±1.79e3 | -1.05e3±1.79e3 | -1.05e3±1.79e3 | -1.55e5±3.00e5 | -1.46e3±2.57e3 | -1.46e3±2.57e3 | -1.46e3±2.57e3 | -1.46e3±2.57e3 | -1.54e5±2.99e5 | -1.60e3±2.83e3 | -1.60e3±2.83e3 | -1.60e3±2.83e3 | -1.60e3±2.83e3 |

Table 32: Overall results for GTOPX unconstrained tasks with 128 solutions and 0th percentile evaluations. In this case, 25% of the values are missing near the worst value and another 25% near the optimal value. Details are the same as Table 5.

| Steps | t = 50 | | | | | t = 100 | | | | | t = 150 | | | | |
|---|---|---|---|---|---|---|---|---|---|---|---|---|---|---|---|
| Task $f(x^*_{DFF})$ | GTOPX 2 −195.44±1.38 | | | | | | | | | | | | | | |
| Metric | FS | SI | OI | SO | SO$_w$ | FS | SI | OI | SO | SO$_w$ | FS | SI | OI | SO | SO$_w$ |
| ARCOO | -805.76±816.52 | -1.84±2.56 | -1.84±2.56 | -1.84±2.56 | -1.84±2.56 | -1.12e3±1.08e3 | -3.10±5.04 | -3.10±5.04 | -3.10±5.04 | -3.10±5.04 | -1.05e3±861.88 | -3.76±5.49 | -3.76±5.49 | -3.76±5.49 | -3.76±5.49 |
| BO | -8.48e3±5.31e3 | -91.71±21.56 | -91.71±21.56 | -91.71±21.56 | -91.71±21.56 | -1.85e4±2.29e4 | -94.05±9.87 | -94.05±9.87 | -94.05±9.87 | -94.05±9.87 | -7.81e3±4.08e3 | -89.88±7.80 | -89.88±7.80 | -89.88±7.80 | -89.88±7.80 |
| CBAS | -195.44±1.38 | 1.00±0.00 | 1.00±0.00 | 1.00±0.00 | 1.00±0.00 | -195.44±1.38 | 1.00±0.00 | 1.00±0.00 | 1.00±0.00 | 1.00±0.00 | -195.44±1.38 | 1.00±0.00 | 1.00±0.00 | 1.00±0.00 | 1.00±0.00 |
| CCDDEA | -222.51±119.44 | -0.37±0.62 | -0.54±1.03 | -0.43±0.76 | -0.46±0.83 | -1.03e3±471.62 | -1.12±1.53 | -1.87±2.54 | -1.39±1.91 | -1.29±1.77 | -1033.43±871.65 | -1.83±2.45 | -3.11±4.06 | -2.29±3.05 | -1.84±2.46 |
| CMAES | -459.36±165.29 | -0.45±0.34 | -0.45±0.34 | -0.45±0.34 | -0.45±0.34 | - | - | - | - | - | - | - | - | - | - |
| TTDDEA | -313.42±57.61 | -7.10±4.69 | -7.50±5.41 | -7.28±5.10 | -7.35±5.14 | -270.37±31.36 | -3.38±2.35 | -3.57±2.69 | -3.47±2.49 | -3.44±2.44 | -264.76±28.84 | -2.10±1.58 | -2.22±1.70 | -2.15±1.67 | -2.10±1.58 |
| Trimentoring | -4.97e4±9.09e4 | -827.57±1.82e3 | -827.57±1.82e3 | -827.57±1.82e3 | -827.57±1.82e3 | -4.01e5±9.69e5 | -1.51e3±2.26e3 | -1.51e3±2.26e3 | -1.51e3±2.26e3 | -1.51e3±2.26e3 | -5.35e5±1.32e6 | -2.01e3±3.17e3 | -2.01e3±3.17e3 | -2.01e3±3.17e3 | -2.01e3±3.17e3 |
| Task $f(x^*_{DFF})$ | GTOPX 3 −151.85±0.66 | | | | | | | | | | | | | | |
| ARCOO | -694.04±640.76 | -2.60±2.96 | -2.60±2.96 | -2.60±2.96 | -2.60±2.96 | -3.72e3±7.66e3 | -8.53±14.70 | -8.53±14.70 | -8.53±14.70 | -8.53±14.70 | -4.77e3±7.92e5 | -15.59±30.30 | -15.59±30.30 | -15.59±30.30 | -15.59±30.30 |
| BO | -5.73e4±3.75e4 | -1.07e3±479.65 | -1.07e3±479.65 | -1.07e3±479.65 | -1.07e3±479.65 | -6.23e4±7.56e4 | -897.52±287.54 | -897.52±287.54 | -897.52±287.54 | -897.52±287.54 | -3.98e4±4.46e4 | -844.31±195.40 | -844.31±195.40 | -844.31±195.40 | -844.31±195.40 |
| CBAS | -151.85±0.66 | 1.00±0.00 | 1.00±0.00 | 1.00±0.00 | 1.00±0.00 | -151.85±0.66 | 1.00±0.00 | 1.00±0.00 | 1.00±0.00 | 1.00±0.00 | -151.85±0.66 | 1.00±0.00 | 1.00±0.00 | 1.00±0.00 | 1.00±0.00 |
| CCDDEA | -60477.94±159157.10 | -8.08±10.38 | -12.24±16.88 | -9.69±12.81 | -10.41±13.93 | -217.58±83.23 | -6.35±11.53 | -10.16±19.25 | -7.80±14.42 | -7.26±13.33 | -215.21±83.37 | -4.13±2.76 | -6.62±12.96 | -5.08±9.71 | -4.14±2.78 |
| CMAES | -341.61±46.39 | -5.11±4.58 | -5.11±4.58 | -5.11±4.58 | -5.11±4.58 | - | - | - | - | - | - | - | - | - | - |
| TTDDEA | -207.69±116.29 | -38.84±31.72 | -46.99±32.48 | -41.84±31.51 | -43.25±31.61 | -1.13e4±1.47e4 | -24.78±20.00 | -30.11±19.10 | -26.75±19.49 | -26.01±19.68 | -1.31e7±3.30e7 | -7.11e3±1.78e4 | -7.26e3±1.78e4 | -7.17e3±1.78e4 | -7.11e3±1.78e4 |
| Trimentoring | -247.62±108.29 | -4.00e3±1.03e4 | -4.00e3±1.03e4 | -4.00e3±1.03e4 | -4.00e3±1.03e4 | -253.11±112.29 | -4.02e3±1.04e4 | -4.02e3±1.04e4 | -4.02e3±1.04e4 | -4.02e3±1.04e4 | -254.54±111.81 | -4.03e3±1.04e4 | -4.03e3±1.04e4 | -4.03e3±1.04e4 | -4.03e3±1.04e4 |
| Task $f(x^*_{DFF})$ | GTOPX 4 −215.74±1.18 | | | | | | | | | | | | | | |
| ARCOO | -1823.17±1724.41 | -6.14±5.86 | -6.14±5.86 | -6.14±5.86 | -6.14±5.86 | -2500.91±1924.49 | -9.53±8.02 | -9.53±8.02 | -9.53±8.02 | -9.53±8.02 | -3387.28±3585.51 | -12.41±10.89 | -12.41±10.89 | -12.41±10.89 | -12.41±10.89 |
| BO | -9.65e5±3.22e6 | -1.13e4±1.68e4 | -1.13e4±1.68e4 | -1.13e4±1.68e4 | -1.13e4±1.68e4 | -1.81e5±1.94e5 | -7.65e3±8.00e3 | -7.65e3±8.00e3 | -7.65e3±8.00e3 | -7.65e3±8.00e3 | -9.28e4±4.02e4 | -6.39e3±5.20e3 | -6.39e3±5.20e3 | -6.39e3±5.20e3 | -6.39e3±5.20e3 |
| CBAS | -215.74±1.18 | 1.00±0.00 | 1.00±0.00 | 1.00±0.00 | 1.00±0.00 | -215.74±1.18 | 1.00±0.00 | 1.00±0.00 | 1.00±0.00 | 1.00±0.00 | -215.74±1.18 | 1.00±0.00 | 1.00±0.00 | 1.00±0.00 | 1.00±0.00 |
| CCDDEA | -512.42±321.32 | -16.34±15.98 | -21.68±21.35 | -18.50±18.20 | -19.42±19.12 | -2.14e5±3.06e5 | -2.11e3±4.60e3 | -2.56e3±5.52e3 | -2.31e3±5.02e3 | -2.24±4.88e3 | -9.80e4±1.61e5 | -1.77e3±3.51e3 | -2.15e3±4.20e3 | -1.94e3±3.83e3 | -1.77e3±3.51e3 |
| CMAES | -698.29±84.07 | -32.08±45.93 | -32.08±45.93 | -32.08±45.93 | -32.08±45.93 | - | - | - | - | - | - | - | - | - | - |
| TTDDEA | -1.24e3±1.90e3 | -584.99±1010.45 | -596.69±1013.27 | -590.52±1001.67 | -592.52±1002.16 | -4.87e3±1.29e4 | -297.17±498.57 | -303.15±499.78 | -299.99±499.08 | -299.04±498.90 | -1.04e3±999.93 | -205.03±329.43 | -209.32±330.94 | -207.06±330.13 | -205.05±329.44 |
| Trimentoring | -2.06e5±5.42e5 | -2.20e3±3.53e3 | -2.20e3±3.53e3 | -2.20e3±3.53e3 | -2.20e3±3.53e3 | -2.14e5±5.64e5 | -2.77e3±4.02e3 | -2.77e3±4.02e3 | -2.77e3±4.02e3 | -2.77e3±4.02e3 | -2.14e5±5.64e5 | -2.97e3±4.29e3 | -2.97e3±4.29e3 | -2.97e3±4.29e3 | -2.97e3±4.29e3 |
| Task $f(x^*_{DFF})$ | GTOPX 6 −112.11±0.33 | | | | | | | | | | | | | | |
| ARCOO | -4.20e3±8.59e3 | -54.87±128.11 | -54.87±128.11 | -54.87±128.11 | -54.87±128.11 | -3.03e4±7.73e4 | -102.86±248.55 | -102.86±248.55 | -102.86±248.55 | -102.86±248.55 | -2.41e5±6.35e5 | -545.01±1412.48 | -545.01±1412.48 | -545.01±1412.48 | -545.01±1412.48 |
| BO | -7.39e3±7.08e3 | -72.58±11.33 | -72.58±11.33 | -72.58±11.33 | -72.58±11.33 | -9.03e3±1.04e4 | -77.27±13.43 | -77.27±13.43 | -77.27±13.43 | -77.27±13.43 | -7.04e3±9.02e3 | -80.04±10.69 | -80.04±10.69 | -80.04±10.69 | -80.04±10.69 |
| CBAS | -112.11±0.33 | 1.00±0.00 | 1.00±0.00 | 1.00±0.00 | 1.00±0.00 | -112.11±0.33 | 1.00±0.00 | 1.00±0.00 | 1.00±0.00 | 1.00±0.00 | -112.11±0.33 | 1.00±0.00 | 1.00±0.00 | 1.00±0.00 | 1.00±0.00 |
| CCDDEA | -210.41±61.68 | -1.30±2.40 | -2.08±4.00 | -1.60±3.00 | -1.73±3.28 | -3.54e3±5354.32 | -11.61±18.56 | -16.73±25.61 | -13.65±21.40 | -12.91±20.37 | -3.54e36.98±535537 | -20.67±34.48 | -29.57±46.87 | -24.23±39.93 | -20.71±34.54 |
| CMAES | -369.34±59.84 | -2.34±0.34 | -2.34±0.34 | -2.34±0.34 | -2.34±0.34 | - | - | - | - | - | - | - | - | - | - |
| TTDDEA | -178.77±72.02 | -7.89±4.21 | -8.43±8.15 | -8.11±8.17 | -8.21±8.16 | -204.06±101.96 | -4.03±4.48 | -4.29±4.43 | -4.13±4.45 | -4.10±4.46 | -217.19±117.42 | -2.82±3.15 | -2.96±3.11 | -2.87±3.13 | -2.82±3.15 |
| Trimentoring | -2.12e4±4.78e4 | -1.11e3±2.13e3 | -1.11e3±2.13e3 | -1.11e3±2.13e3 | -1.11e3±2.13e3 | -2.09e4±4.71e4 | -1.13e3±2.15e3 | -1.13e3±2.15e3 | -1.13e3±2.15e3 | -1.13e3±2.15e3 | -2.10e4±4.69e4 | -1.13e3±2.15e3 | -1.13e3±2.15e3 | -1.13e3±2.15e3 | -1.13e3±2.15e3 |

Table 33: Overall results for GTOPX unconstrained tasks with 128 solutions and 0th percentile evaluations. In this case, 30% of the values are missing near the worst value and another 20% near the optimal value. Details are the same as Table 5.

| Steps | t = 50 | | | | | t = 100 | | | | | t = 150 | | | | |
|---|---|---|---|---|---|---|---|---|---|---|---|---|---|---|---|
| Task $f(x^*_{DFF})$ | GTOPX 2 −175.10±0.75 | | | | | | | | | | | | | | |
| Metric | FS | SI | OI | SO | SO$_w$ | FS | SI | OI | SO | SO$_w$ | FS | SI | OI | SO | SO$_w$ |
| ARCOO | -827.53±696.69 | -3.08±2.35 | -3.08±2.35 | -3.08±2.35 | -3.08±2.35 | -1.51e3±1.92e3 | -5.61±5.95 | -5.61±5.95 | -5.61±5.95 | -5.61±5.95 | -1.85e3±2.01e3 | -7.93±9.71 | -7.93±9.71 | -7.93±9.71 | -7.93±9.71 |
| BO | -1.08e4±7.46e3 | -105.06±28.57 | -105.06±28.57 | -105.06±28.57 | -105.06±28.57 | -8.05e3±6.95e3 | -100.38±20.77 | -100.38±20.77 | -100.38±20.77 | -100.38±20.77 | -1.54e4±1.34e4 | -97.62±15.82 | -97.62±15.82 | -97.62±15.82 | -97.62±15.82 |
| CBAS | -175.10±0.75 | 1.00±0.00 | 1.00±0.00 | 1.00±0.00 | 1.00±0.00 | -175.10±0.75 | 1.00±0.00 | 1.00±0.00 | 1.00±0.00 | 1.00±0.00 | -175.10±0.75 | 1.00±0.00 | 1.00±0.00 | 1.00±0.00 | 1.00±0.00 |
| CCDDEA | -187.65±28.51 | -0.68±1.24 | -1.19±2.24 | -0.87±1.59 | -0.95±1.77 | -1.33e3±1.80e3 | -1.89±2.82 | -3.02±4.38 | -2.32±3.43 | -2.16±3.23 | -1.55e3±2.13e3 | -3.56±5.92 | -5.63±9.19 | -4.35±7.20 | -3.56±5.94 |
| CMAES | -404.68±107.23 | -0.78±0.46 | -0.78±0.46 | -0.78±0.46 | -0.78±0.46 | - | - | - | - | - | - | - | - | - | - |
| TTDDEA | -311.00±71.27 | -6.61±4.33 | -6.61±4.33 | -6.61±4.33 | -6.61±4.33 | -278.62±39.60 | -3.29±2.20 | -3.29±2.20 | -3.29±2.20 | -3.29±2.20 | -277.25±31.83 | -2.17±1.51 | -2.17±1.51 | -2.17±1.51 | -2.17±1.51 |
| Trimentoring | -4.95e5±1.11e6 | -4.46e3±9.08e3 | -4.46e3±9.08e3 | -4.46e3±9.08e3 | -4.46e3±9.08e3 | -4.41e5±9.73e5 | -4.24e3±8.68e3 | -4.24e3±8.68e3 | -4.24e3±8.68e3 | -4.24e3±8.68e3 | -4.29e5±9.50e5 | -4.13e3±8.46e3 | -4.13e3±8.46e3 | -4.13e3±8.46e3 | -4.13e3±8.46e3 |
| Task $f(x^*_{DFF})$ | GTOPX 3 −134.83±0.56 | | | | | | | | | | | | | | |
| ARCOO | -970.27±849.88 | -2.32±1.81 | -2.32±1.81 | -2.32±1.81 | -2.32±1.81 | -2.54e3±3.03e3 | -8.22±7.85 | -8.22±7.85 | -8.22±7.85 | -8.22±7.85 | -5.46e3±7.70e3 | -19.34±24.21 | -19.34±24.21 | -19.34±24.21 | -19.34±24.21 |
| BO | -3.35e5±6.49e5 | -669.30±166.16 | -669.30±166.16 | -669.30±166.16 | -669.30±166.16 | -4.53e5±1.09e6 | -1.03e3±563.84 | -1.03e3±563.84 | -1.03e3±563.84 | -1.03e3±563.84 | -3.50e4±3.08e4 | -1.16e3±525.11 | -1.16e3±525.11 | -1.16e3±525.11 | -1.16e3±525.11 |
| CBAS | -134.83±0.56 | 1.00±0.00 | 1.00±0.00 | 1.00±0.00 | 1.00±0.00 | -134.83±0.56 | 1.00±0.00 | 1.00±0.00 | 1.00±0.00 | 1.00±0.00 | -134.83±0.56 | 1.00±0.00 | 1.00±0.00 | 1.00±0.00 | 1.00±0.00 |
| CCDDEA | -185.39±36.98 | -6.68±11.83 | -11.76±22.17 | -8.50±15.45 | -9.37±17.17 | -323.90±296.82 | -3.60±6.24 | -6.37±11.75 | -4.59±8.15 | -4.21±7.91 | -323.79±296.82 | -5.02±9.30 | -4.72±9.08 | -3.39±6.28 | -2.66±4.81 |
| CMAES | -308.13±33.90 | -9.20±6.26 | -9.20±6.26 | -9.20±6.26 | -9.20±6.26 | - | - | - | - | - | - | - | - | - | - |
| TTDDEA | -334.87±285.08 | -38.59±17.66 | -46.71±23.91 | -41.86±19.74 | -43.27±20.83 | -2.69e4±3.64e4 | -47.16±42.51 | -53.70±47.40 | -49.91±44.58 | -48.92±43.84 | -1.53e7±3.43e7 | -1.92e4±4.48e4 | -1.92e4±4.48e4 | -1.92e4±4.48e4 | -1.92e4±4.48e4 |
| Trimentoring | -9.67e7±1.17e8 | -1.11e6±2.84e6 | -1.11e6±2.84e6 | -1.11e6±2.84e6 | -1.11e6±2.84e6 | -1.63e8±3.08e8 | -1.29e6±2.76e6 | -1.29e6±2.76e6 | -1.29e6±2.76e6 | -1.29e6±2.76e6 | -2.32e8±4.66e8 | -1.48e6±3.00e6 | -1.48e6±3.00e6 | -1.48e6±3.00e6 | -1.48e6±3.00e6 |
| Task $f(x^*_{DFF})$ | GTOPX 4 −194.03±0.88 | | | | | | | | | | | | | | |
| ARCOO | -5.44e4±1.15e5 | -89.34±178.14 | -89.34±178.14 | -89.34±178.14 | -89.34±178.14 | -2.63e5±2.67e5 | -661.55±1.57e3 | -661.55±1.57e3 | -661.55±1.57e3 | -661.55±1.57e3 | -5.13e5±1.33e6 | -1.21e3±3.01e3 | -1.21e3±3.01e3 | -1.21e3±3.01e3 | -1.21e3±3.01e3 |
| BO | -2.49e5±5.18e5 | -4.30e3±1.81e3 | -4.30e3±1.81e3 | -4.30e3±1.81e3 | -4.30e3±1.81e3 | -4.79e5±4.95e5 | -5.19e3±2.44e3 | -5.19e3±2.44e3 | -5.19e3±2.44e3 | -5.19e3±2.44e3 | -1.57e5±1.96e5 | -4.61e3±1.60e3 | -4.61e3±1.60e3 | -4.61e3±1.60e3 | -4.61e3±1.60e3 |
| CBAS | -194.03±0.88 | 1.00±0.00 | 1.00±0.00 | 1.00±0.00 | 1.00±0.00 | -194.03±0.88 | 1.00±0.00 | 1.00±0.00 | 1.00±0.00 | 1.00±0.00 | -194.03±0.88 | 1.00±0.00 | 1.00±0.00 | 1.00±0.00 | 1.00±0.00 |
| CCDDEA | -494.13±435.49 | -78.23±177.02 | -97.09±213.46 | -86.51±193.99 | -89.77±199.88 | -2.04e5±5.47e5 | -473.11±942.46 | -763.77±1.48e3 | -580.78±1.07e3 | -540.29±984.96 | -3.22e6±7.84e6 | -6.32e3±1.45e4 | -8.03e3±1.75e4 | -7.05e3±1.58e4 | -6.33e3±1.45e4 |
| CMAES | -696.85±97.16 | -18.31±10.99 | -18.31±10.99 | -18.31±10.99 | -18.31±10.99 | - | - | - | - | - | - | - | - | - | - |
| TTDDEA | -355.30±75.60 | -627.84±887.93 | -627.84±887.93 | -627.84±887.93 | -627.84±887.93 | -410.44±83.95 | -311.03±439.43 | -311.03±439.43 | -311.03±439.43 | -311.03±439.43 | -394.15±183.45 | -206.41±292.28 | -206.86±292.00 | -206.62±292.15 | -206.41±292.27 |
| Trimentoring | -619.08±247.53 | -581.09±1.47e3 | -581.09±1.47e3 | -581.09±1.47e3 | -581.09±1.47e3 | -664.22±286.44 | -584.21±1.47e3 | -584.21±1.47e3 | -584.21±1.47e3 | -584.21±1.47e3 | -590.88±221.70 | -585.19±1.48e3 | -585.19±1.48e3 | -585.19±1.48e3 | -585.19±1.48e3 |
| Task $f(x^*_{DFF})$ | GTOPX 6 −102.96±0.29 | | | | | | | | | | | | | | |
| ARCOO | -5.28e3±1.25e4 | -24.48±50.22 | -24.48±50.22 | -24.48±50.22 | -24.48±50.22 | -6.09e4±1.37e5 | -262.86±668.60 | -262.86±668.60 | -262.86±668.60 | -262.86±668.60 | -7.01e4±1.80e5 | -593.93±1.55e3 | -593.93±1.55e3 | -593.93±1.55e3 | -593.93±1.55e3 |
| BO | -5.69e3±3.96e3 | -84.65±15.44 | -84.65±15.44 | -84.65±15.44 | -84.65±15.44 | -4.63e3±2.46e3 | -88.86±15.22 | -88.86±15.22 | -88.86±15.22 | -88.86±15.22 | -5.70e3±6.75e3 | -88.53±11.68 | -88.53±11.68 | -88.53±11.68 | -88.53±11.68 |
| CBAS | -102.96±0.29 | 1.00±0.00 | 1.00±0.00 | 1.00±0.00 | 1.00±0.00 | -102.96±0.29 | 1.00±0.00 | 1.00±0.00 | 1.00±0.00 | 1.00±0.00 | -102.96±0.29 | 1.00±0.00 | 1.00±0.00 | 1.00±0.00 | 1.00±0.00 |
| CCDDEA | -178.28±64.61 | -0.91±0.50 | -1.26±0.76 | -1.05±0.60 | -1.12±0.69 | -1.15e3±1.73e3 | -5.57±9.57 | -7.42±12.57 | -6.36±10.87 | -6.30±10.41 | -1.37e3±1.71e3 | -8.34±12.91 | -11.09±16.88 | -9.51±14.63 | -8.35±12.93 |
| CMAES | -330.65±35.89 | -3.68±1.66 | -3.68±1.66 | -3.68±1.66 | -3.68±1.66 | - | - | - | - | - | - | - | - | - | - |
| TTDDEA | -277.33±131.60 | -13.27±9.24 | -13.37±9.16 | -13.32±9.20 | -13.33±9.19 | -176.81±80.17 | -6.70±4.72 | -6.85±4.63 | -6.77±4.68 | -6.75±4.69 | -178.32±55.62 | -4.20±2.68 | -4.63±3.18 | -4.39±2.87 | -4.20±2.68 |
| Trimentoring | -2.34e5±5.49e5 | -7.33e3±1.36e4 | -7.33e3±1.36e4 | -7.33e3±1.36e4 | -7.33e3±1.36e4 | -4.60e5±1.14e6 | -7.36e3±1.52e4 | -7.36e3±1.52e4 | -7.36e3±1.52e4 | -7.36e3±1.52e4 | -1.09e5±2.24e5 | -6.80e3±1.43e4 | -6.80e3±1.43e4 | -6.80e3±1.43e4 | -6.80e3±1.43e4 |

Table 34: Overall results for GTOPX unconstrained tasks with 128 solutions and 0th percentile evaluations. In this case, 40% of the values are missing near the worst value and another 10% near the optimal value. Details are the same as Table 5.

| Steps | | t = 50 | | | | | t = 100 | | | | | t = 150 | | | |
|---|---|---|---|---|---|---|---|---|---|---|---|---|---|---|---|
| Task | | | | | | | GTOPX 2 | | | | | | | | |
| $f(x^*_{\text{OFF}})$ | | | | | | | -136.28±0.39 | | | | | | | | |
| Metric | FS | SI | OI | SO | SO_ω | FS | SI | OI | SO | SO_ω | FS | SI | OI | SO | SO_ω |
| ARCOO | -655.75±632.56 | -3.80±2.67 | -3.80±2.67 | -3.80±2.67 | -3.80±2.67 | -795.84±812.60 | -4.98±5.37 | -4.98±5.37 | -4.98±5.37 | -4.98±5.37 | -1.04e3±923.98 | -5.95±6.54 | -5.95±6.54 | -5.95±6.54 | -5.95±6.54 |
| BO | -6.53e3±2.06e3 | -140.68±17.47 | -140.68±17.47 | -140.68±17.47 | -140.68±17.47 | -8.74e3±5.53e3 | -141.29±19.35 | -141.29±19.35 | -141.29±19.35 | -141.29±19.35 | -7.00e3±3.58e3 | -139.87±19.19 | -139.87±19.19 | -139.87±19.19 | -139.87±19.19 |
| CBAS | -136.28±0.39 | 1.00±0.00 | 1.00±0.00 | 1.00±0.00 | 1.00±0.00 | -136.28±0.39 | 1.00±0.00 | 1.00±0.00 | 1.00±0.00 | 1.00±0.00 | -136.28±0.39 | 1.00±0.00 | 1.00±0.00 | 1.00±0.00 | 1.00±0.00 |
| CCDDEA | -252.21±197.90 | -1.97±3.39 | -3.24±3.79 | -2.44±4.27 | -2.67±4.68 | -1.42e4±5.06e4 | -26.78±57.25 | -45.65±98.64 | -33.75±72.45 | -31.12±66.70 | -6.01e4±1.50e5 | -132.53±322.07 | -227.52±554.16 | -167.47±407.39 | -132.89±322.97 |
| CMAES | -358.75±117.91 | -2.30±0.74 | -2.30±0.74 | -2.30±0.74 | -2.30±0.74 | -362.37±90.77 | -8.01±4.70 | -8.01±4.70 | -8.01±4.70 | -8.01±4.70 | -384.09±112.04 | -6.29±3.52 | -6.29±3.52 | -6.29±3.52 | -6.29±3.52 |
| TTDDEA | -392.64±94.00 | -13.15±8.47 | -13.15±8.47 | -13.15±8.47 | -13.15±8.47 | - | - | - | - | - | - | - | - | - | - |
| Trimentoring | -793.73±1.55e3 | -494.19±1.07e3 | -494.19±1.07e3 | -494.19±1.07e3 | -494.19±1.07e3 | -3.75e5±9.89e5 | -1.39e3±2.39e3 | -1.39e3±2.39e3 | -1.39e3±2.39e3 | -1.39e3±2.39e3 | -4.01e5±1.06e6 | -2.60e3±5.50e3 | -2.60e3±5.50e3 | -2.60e3±5.50e3 | -2.60e3±5.50e3 |
| Task | | | | | | | GTOPX 3 | | | | | | | | |
| $f(x^*_{\text{OFF}})$ | | | | | | | -102.69±0.49 | | | | | | | | |
| ARCOO | -1.27e3±1.64e3 | -7.40±7.40 | -7.40±7.40 | -7.40±7.40 | -7.40±7.40 | -5.54e4±1.38e5 | -129.54±308.05 | -129.54±308.05 | -129.54±308.05 | -129.54±308.05 | -1.07e5±2.58e5 | -533.87±1.32e3 | -533.87±1.32e3 | -533.87±1.32e3 | -533.87±1.32e3 |
| BO | -3.96e4±3.45e4 | -1.41e3±699.47 | -1.41e3±699.47 | -1.41e3±699.47 | -1.41e3±699.47 | -2.28e4±5.57e3 | -1.44e3±436.10 | -1.44e3±436.10 | -1.44e3±436.10 | -1.44e3±436.10 | -4.96e3±4.16e4 | -1.46e3±487.99 | -1.46e3±487.99 | -1.46e3±487.99 | -1.46e3±487.99 |
| CBAS | -102.69±0.49 | 1.00±0.00 | 1.00±0.00 | 1.00±0.00 | 1.00±0.00 | -102.69±0.49 | 1.00±0.00 | 1.00±0.00 | 1.00±0.00 | 1.00±0.00 | -102.69±0.49 | 1.00±0.00 | 1.00±0.00 | 1.00±0.00 | 1.00±0.00 |
| CCDDEA | -218.32±119.85 | -71.73±181.55 | -122.58±131.23 | -90.48±229.61 | -99.25±292.15 | -2.52e3±4.37e3 | -39.93±85.13 | -72.94±162.41 | -51.48±111.70 | -47.05±101.43 | -2.57e3±4.34e3 | -34.45±61.73 | -60.68±115.35 | -43.78±80.51 | -34.54±61.92 |
| CMAES | -374.72±79.25 | -15.87±11.22 | -15.87±11.22 | -15.87±11.22 | -15.87±11.22 | - | - | - | - | - | - | - | - | - | - |
| TTDDEA | -220.51±239.99 | -81.45±162.63 | -100.89±185.07 | -89.83±72.00 | -93.21±75.91 | -2.68e5±7.06e5 | -2.77e3±7.22e3 | -2.78e3±7.21e3 | -2.77e3±7.22e3 | -2.77e3±7.22e3 | -9.42e5±1.72e6 | -1.15e4±2.74e4 | -1.15e4±2.74e4 | -1.15e4±2.74e4 | -1.15e4±2.74e4 |
| Trimentoring | -3.94e5±1.01e6 | -3.37e4±8.30e4 | -3.37e4±8.30e4 | -3.37e4±8.30e4 | -3.37e4±8.30e4 | -1.13e8±2.97e8 | -3.40e5±8.33e5 | -3.40e5±8.33e5 | -3.40e5±8.33e5 | -3.40e5±8.33e5 | -1.01e8±2.59e8 | -6.24e5±1.57e6 | -6.24e5±1.57e6 | -6.24e5±1.57e6 | -6.24e5±1.57e6 |
| Task | | | | | | | GTOPX 4 | | | | | | | | |
| $f(x^*_{\text{OFF}})$ | | | | | | | -153.62±0.60 | | | | | | | | |
| ARCOO | -5.26e3±1.08e4 | -16.22±28.66 | -16.22±28.66 | -16.22±28.66 | -16.22±28.66 | -1.01e5±2.26e5 | -157.91±326.44 | -157.91±326.44 | -157.91±326.44 | -157.91±326.44 | -1.09e7±2.81e7 | -7.89e3±2.03e4 | -7.89e3±2.03e4 | -7.89e3±2.03e4 | -7.89e3±2.03e4 |
| BO | -4.28e5±3.87e5 | -8.41e3±6.61e3 | -8.41e3±6.61e3 | -8.41e3±6.61e3 | -8.41e3±6.61e3 | -2.72e5±2.93e5 | -6.29e3±3.14e3 | -6.29e3±3.14e3 | -6.29e3±3.14e3 | -6.29e3±3.14e3 | -2.59e5±2.60e5 | -6.20e3±1.96e3 | -6.20e3±1.96e3 | -6.20e3±1.96e3 | -6.20e3±1.96e3 |
| CBAS | -153.62±0.60 | 1.00±0.00 | 1.00±0.00 | 1.00±0.00 | 1.00±0.00 | -153.62±0.60 | 1.00±0.00 | 1.00±0.00 | 1.00±0.00 | 1.00±0.00 | -153.62±0.60 | 1.00±0.00 | 1.00±0.00 | 1.00±0.00 | 1.00±0.00 |
| CCDDEA | -2.79e3±6.08e3 | -1.48e3±3.43e3 | -2.01e3±4.54e3 | -1.70e3±3.91e3 | -1.79e3±4.38e3 | -6.28e5±1.52e6 | -2.04e5±5.34e5 | -2.70e5±7.07e5 | -2.32e5±6.08e5 | -2.22e5±5.82e5 | -6.30e5±1.51e6 | -1.38e5±3.59e5 | -1.82e5±4.76e5 | -1.57e5±4.09e5 | -1.38e5±3.60e5 |
| CMAES | -682.92±71.60 | -96.34±149.12 | -96.34±149.12 | -96.34±149.12 | -96.34±149.12 | - | - | - | - | - | - | - | - | - | - |
| TTDDEA | -350.19±106.34 | -446.07±585.39 | -446.07±585.39 | -446.07±585.39 | -446.07±585.39 | -430.28±148.52 | -218.98±281.68 | -222.00±289.43 | -220.47±285.50 | -219.98±284.25 | -440.52±114.41 | -138.44±166.77 | -148.35±192.08 | -143.07±178.56 | -138.50±166.91 |
| Trimentoring | -607.72±504.22 | -84.93±104.42 | -84.93±104.42 | -84.93±104.42 | -84.93±104.42 | -421.67±266.93 | -85.39±104.95 | -85.39±104.95 | -85.39±104.95 | -85.39±104.95 | -481.59±329.60 | -85.52±105.14 | -85.52±105.14 | -85.52±105.14 | -85.52±105.14 |
| Task | | | | | | | GTOPX 6 | | | | | | | | |
| $f(x^*_{\text{OFF}})$ | | | | | | | -83.37±0.23 | | | | | | | | |
| ARCOO | -3.66e3±7.74e3 | -31.17±53.58 | -31.17±53.58 | -31.17±53.58 | -31.17±53.58 | -2.17e5±5.70e5 | -1.16e3±3.02e3 | -1.16e3±3.02e3 | -1.16e3±3.02e3 | -1.16e3±3.02e3 | -1.40e5±3.67e5 | -1.76e3±4.62e3 | -1.76e3±4.62e3 | -1.76e3±4.62e3 | -1.76e3±4.62e3 |
| BO | -5.84e3±3.07e3 | -152.84±118.70 | -152.84±118.70 | -152.84±118.70 | -152.84±118.70 | -4.94e3±2.84e3 | -143.94±62.37 | -143.94±62.37 | -143.94±62.37 | -143.94±62.37 | -7.94e3±4.46e3 | -139.99±44.07 | -139.99±44.07 | -139.99±44.07 | -139.99±44.07 |
| CBAS | -83.37±0.23 | 1.00±0.00 | 1.00±0.00 | 1.00±0.00 | 1.00±0.00 | -83.37±0.23 | 1.00±0.00 | 1.00±0.00 | 1.00±0.00 | 1.00±0.00 | -83.37±0.23 | 1.00±0.00 | 1.00±0.00 | 1.00±0.00 | 1.00±0.00 |
| CCDDEA | -243.40±193.14 | -4.05±3.85 | -5.04±4.20 | -4.44±4.41 | -4.61±4.07 | -7.11e3±1.04e4 | -71.19±120.90 | -113.19±211.99 | -86.39±152.38 | -80.63±140.22 | -7.12e3±1.06e4 | -84.75±142.46 | -119.54±194.01 | -97.93±161.20 | -84.89±142.66 |
| CMAES | -383.83±101.84 | -5.69±0.73 | -5.69±0.73 | -5.69±0.73 | -5.69±0.73 | - | - | - | - | - | - | - | - | - | - |
| TTDDEA | -2.94e3±5.91e3 | -90.77±136.01 | -90.77±136.01 | -90.77±136.01 | -90.77±136.01 | -6.54e3±1.11e4 | -75.95±116.83 | -75.95±116.83 | -75.95±116.83 | -75.95±116.83 | -9.07e3±2.12e4 | -96.17±176.03 | -96.17±176.03 | -96.17±176.03 | -96.17±176.03 |
| Trimentoring | -219.40±94.17 | -843.36±2.11e3 | -843.36±2.11e3 | -843.36±2.11e3 | -843.36±2.11e3 | -216.31±91.25 | -844.89±2.12e3 | -844.89±2.12e3 | -844.89±2.12e3 | -844.89±2.12e3 | -250.16±147.22 | -845.29±2.12e3 | -845.29±2.12e3 | -845.29±2.12e3 | -845.29±2.12e3 |

Table 35: Overall results for mujoco unconstrained tasks with 128 solutions and 100th percentile evaluations. In this case, 0% of the values are missing near the worst value and another 50% near the optimal value. Details are the same as Table 5.

| Steps | | t = 50 | | | | | t = 100 | | | | | t = 150 | | | |
|---|---|---|---|---|---|---|---|---|---|---|---|---|---|---|---|
| Task | | | | | | | Mujoco 1 | | | | | | | | |
| $f(x^*_{\text{OFF}})$ | | | | | | | 0.37±0.00 | | | | | | | | |
| Metric | FS | SI | OI | SO | SO_ω | FS | SI | OI | SO | SO_ω | FS | SI | OI | SO | SO_ω |
| ARCOO | 0.39±0.05 | 0.94±0.12 | 1.00±0.01 | 0.97±0.07 | 0.98±0.05 | 0.64±0.05 | 0.69±0.04 | 1.26±0.10 | 0.89±0.05 | 0.82±0.04 | 0.72±0.12 | 0.73±0.04 | 1.45±0.08 | 0.97±0.06 | 0.73±0.04 |
| BO | 0.81±0.13 | 0.85±0.02 | 1.68±0.02 | 1.13±0.02 | 1.27±0.02 | 0.72±0.11 | 0.85±0.01 | 1.68±0.02 | 1.12±0.01 | 1.02±0.00 | 0.88±0.05 | 0.84±0.01 | 1.68±0.02 | 1.12±0.01 | 0.84±0.02 |
| CBAS | 0.00±0.00 | -35.22±0.12 | -35.22±0.12 | -35.22±0.12 | -35.22±0.12 | 0.00±0.00 | -35.41±0.12 | -35.41±0.12 | -35.41±0.12 | -35.41±0.12 | 0.00±0.00 | -35.47±0.12 | -35.47±0.12 | -35.47±0.12 | -35.47±0.12 |
| CCDDEA | 0.75±0.09 | 0.79±0.03 | 1.58±0.05 | 1.06±0.04 | 1.19±0.04 | 0.78±0.07 | 0.83±0.04 | 1.67±0.07 | 1.11±0.05 | 1.00±0.04 | 0.78±0.07 | 0.85±0.05 | 1.70±0.09 | 1.13±0.06 | 0.85±0.05 |
| CMAES | 0.97±0.01 | 0.98±0.00 | 1.96±0.01 | 1.31±0.00 | 1.48±0.01 | - | - | - | - | - | - | - | - | - | - |
| TTDDEA | 0.25±0.10 | 0.45±0.09 | 0.89±0.17 | 0.59±0.11 | 0.67±0.13 | 0.22±0.09 | 0.37±0.10 | 0.74±0.20 | 0.49±0.13 | 0.45±0.12 | 0.23±0.09 | 0.34±0.11 | 0.69±0.22 | 0.46±0.14 | 0.34±0.11 |
| Trimentoring | 0.91±0.07 | 0.92±0.03 | 1.84±0.05 | 1.23±0.03 | 1.38±0.04 | 0.89±0.06 | 0.94±0.03 | 1.87±0.07 | 1.25±0.04 | 1.12±0.04 | 0.88±0.04 | 0.93±0.04 | 1.87±0.07 | 1.25±0.05 | 0.94±0.04 |
| Task | | | | | | | Mujoco 2 | | | | | | | | |
| $f(x^*_{\text{OFF}})$ | | | | | | | -0.10±0.00 | | | | | | | | |
| ARCOO | -0.03±0.17 | 0.95±0.13 | 1.01±0.02 | 0.97±0.07 | 0.98±0.05 | 0.42±0.09 | 0.64±0.04 | 1.23±0.09 | 0.84±0.04 | 0.76±0.04 | 0.43±0.09 | 0.73±0.03 | 1.47±0.06 | 0.98±0.04 | 0.73±0.03 |
| BO | 0.45±0.12 | 0.78±0.07 | 1.44±0.13 | 1.01±0.08 | 1.12±0.08 | 0.53±0.18 | 0.72±0.07 | 1.44±0.13 | 0.96±0.09 | 0.87±0.08 | 0.50±0.18 | 0.72±0.06 | 1.44±0.13 | 0.96±0.09 | 0.72±0.06 |
| CBAS | -1.86±0.00 | -173.04±0.27 | -173.04±0.27 | -173.04±0.27 | -173.04±0.27 | -1.86±0.00 | -173.95±0.27 | -173.95±0.27 | -173.95±0.27 | -173.95±0.27 | -1.86±0.00 | -174.25±0.27 | -174.25±0.27 | -174.25±0.27 | -174.25±0.27 |
| CCDDEA | 0.11±0.26 | 0.75±0.04 | 1.49±0.08 | 1.00±0.05 | 1.12±0.06 | 0.25±0.11 | 0.75±0.05 | 1.50±0.10 | 1.00±0.07 | 0.90±0.06 | 0.25±0.11 | 0.76±0.06 | 1.52±0.13 | 1.01±0.09 | 0.76±0.06 |
| CMAES | 0.59±0.05 | 0.90±0.03 | 1.80±0.06 | 1.20±0.04 | 1.35±0.04 | - | - | - | - | - | - | - | - | - | - |
| TTDDEA | 0.17±0.43 | 0.64±0.07 | 1.29±0.15 | 0.86±0.10 | 0.97±0.11 | 0.23±0.41 | 0.61±0.08 | 1.23±0.16 | 0.82±0.10 | 0.74±0.10 | 0.24±0.49 | 0.61±0.08 | 1.22±0.16 | 0.81±0.11 | 0.61±0.08 |
| Trimentoring | 0.49±0.24 | 0.85±0.05 | 1.68±0.10 | 1.12±0.07 | 1.27±0.07 | 0.40±0.13 | 0.85±0.06 | 1.68±0.11 | 1.13±0.08 | 1.02±0.07 | 0.40±0.14 | 0.83±0.06 | 1.67±0.12 | 1.11±0.08 | 0.84±0.04 |

Table 36: Overall results for mujoco unconstrained tasks with 128 solutions and 100th percentile evaluations. In this case, 0% of the values are missing near the worst value and another 40% near the optimal value. Details are the same as Table 5.

| Steps | | t = 50 | | | | | t = 100 | | | | | t = 150 | | | |
|---|---|---|---|---|---|---|---|---|---|---|---|---|---|---|---|
| Task | | | | | | | Mujoco 1 | | | | | | | | |
| $f(x^*_{\text{OFF}})$ | | | | | | | 0.40±0.00 | | | | | | | | |
| Metric | FS | SI | OI | SO | SO_ω | FS | SI | OI | SO | SO_ω | FS | SI | OI | SO | SO_ω |
| ARCOO | 0.40±0.00 | 1.00±0.00 | 1.00±0.00 | 1.00±0.00 | 1.00±0.00 | 0.68±0.15 | 0.72±0.12 | 1.13±0.10 | 0.87±0.07 | 0.82±0.09 | 0.78±0.10 | 0.68±0.05 | 1.37±0.09 | 0.91±0.06 | 0.69±0.05 |
| BO | 0.80±0.10 | 0.84±0.01 | 1.66±0.02 | 1.11±0.02 | 1.25±0.02 | 0.82±0.08 | 0.83±0.01 | 1.67±0.01 | 1.11±0.01 | 1.00±0.01 | 0.72±0.09 | 0.83±0.01 | 1.66±0.01 | 1.11±0.01 | 0.83±0.01 |
| CBAS | 0.00±0.00 | -38.87±0.22 | -38.87±0.22 | -38.87±0.22 | -38.87±0.22 | 0.00±0.00 | -39.08±0.22 | -39.08±0.22 | -39.08±0.22 | -39.08±0.22 | 0.00±0.00 | -39.15±0.22 | -39.15±0.22 | -39.15±0.22 | -39.15±0.22 |
| CCDDEA | 0.73±0.16 | 0.77±0.07 | 1.54±0.15 | 1.03±0.10 | 1.16±0.11 | 0.77±0.15 | 0.82±0.07 | 1.63±0.15 | 1.09±0.10 | 0.98±0.09 | 0.77±0.15 | 0.83±0.09 | 1.66±0.17 | 1.11±0.12 | 0.83±0.09 |
| CMAES | 0.97±0.02 | 0.98±0.01 | 1.97±0.01 | 1.31±0.01 | 1.48±0.00 | - | - | - | - | - | - | - | - | - | - |
| TTDDEA | 0.22±0.10 | 0.41±0.11 | 0.83±0.22 | 0.55±0.14 | 0.62±0.16 | 0.22±0.10 | 0.33±0.12 | 0.67±0.23 | 0.45±0.15 | 0.40±0.14 | 0.22±0.10 | 0.30±0.12 | 0.60±0.25 | 0.40±0.16 | 0.30±0.12 |
| Trimentoring | 0.86±0.18 | 0.93±0.03 | 1.72±0.28 | 1.19±0.08 | 1.33±0.13 | 0.84±0.17 | 0.94±0.03 | 1.76±0.29 | 1.22±0.09 | 1.11±0.05 | 0.86±0.18 | 0.95±0.03 | 1.77±0.30 | 1.23±0.09 | 0.95±0.03 |
| Task | | | | | | | Mujoco 2 | | | | | | | | |
| $f(x^*_{\text{OFF}})$ | | | | | | | -0.04±0.00 | | | | | | | | |
| ARCOO | -0.04±0.00 | 1.00±0.00 | 1.00±0.00 | 1.00±0.00 | 1.00±0.00 | 0.40±0.17 | 0.68±0.06 | 1.19±0.06 | 0.86±0.05 | 0.80±0.05 | 0.48±0.18 | 0.71±0.03 | 1.42±0.05 | 0.94±0.03 | 0.71±0.03 |
| BO | 0.44±0.25 | 0.80±0.04 | 1.41±0.08 | 1.02±0.04 | 1.12±0.04 | 0.42±0.12 | 0.73±0.06 | 1.41±0.07 | 0.96±0.06 | 0.87±0.06 | 0.44±0.09 | 0.71±0.04 | 1.42±0.08 | 0.95±0.05 | 0.71±0.04 |
| CBAS | -1.86±0.00 | -179.29±0.34 | -179.29±0.34 | -179.29±0.34 | -179.29±0.34 | -1.86±0.00 | -180.23±0.34 | -180.23±0.34 | -180.23±0.34 | -180.23±0.34 | -1.86±0.00 | -180.54±0.34 | -180.54±0.34 | -180.54±0.34 | -180.54±0.34 |
| CCDDEA | 0.15±0.16 | 0.75±0.04 | 1.49±0.09 | 1.00±0.06 | 1.12±0.07 | 0.32±0.12 | 0.76±0.06 | 1.51±0.12 | 1.01±0.08 | 0.91±0.07 | 0.32±0.12 | 0.77±0.08 | 1.55±0.15 | 1.03±0.10 | 0.78±0.08 |
| CMAES | 0.62±0.08 | 0.87±0.04 | 1.74±0.09 | 1.16±0.06 | 1.31±0.07 | - | - | - | - | - | - | - | - | - | - |
| TTDDEA | -0.05±0.70 | 0.61±0.08 | 1.23±0.16 | 0.82±0.11 | 0.93±0.12 | -0.04±0.21 | 0.58±0.07 | 1.16±0.15 | 0.77±0.10 | 0.70±0.08 | 0.04±0.22 | 0.57±0.08 | 1.15±0.15 | 0.76±0.10 | 0.57±0.08 |
| Trimentoring | 0.32±0.16 | 0.86±0.06 | 1.58±0.23 | 1.10±0.06 | 1.23±0.10 | 0.29±0.14 | 0.86±0.07 | 1.57±0.24 | 1.10±0.07 | 1.01±0.06 | 0.36±0.24 | 0.85±0.07 | 1.58±0.24 | 1.09±0.07 | 0.85±0.07 |

Table 37: Overall results for mujoco unconstrained tasks with 128 solutions and 100th percentile evaluations. In this case, 0% of the values are missing near the worst value and another 30% near the optimal value. Details are the same as Table 5.

| Steps | t = 50 | | | | | t = 100 | | | | | t = 150 | | | | |
|---|---|---|---|---|---|---|---|---|---|---|---|---|---|---|---|
| Task $f(x^*_{OFF})$ | Mujoco 1 $0.45_{\pm0.00}$ | | | | | | | | | | | | | | |
| Metric | FS | SI | OI | SO | SO$_\omega$ | FS | SI | OI | SO | SO$_\omega$ | FS | SI | OI | SO | SO$_\omega$ |
| ARCOO | $0.45_{\pm0.00}$ | $1.00_{\pm0.00}$ | $1.00_{\pm0.00}$ | $1.00_{\pm0.00}$ | $1.00_{\pm0.00}$ | $0.63_{\pm0.12}$ | $0.74_{\pm0.14}$ | $1.09_{\pm0.08}$ | $0.87_{\pm0.09}$ | $0.82_{\pm0.11}$ | $0.74_{\pm0.12}$ | $0.67_{\pm0.07}$ | $1.31_{\pm0.11}$ | $0.89_{\pm0.08}$ | $0.67_{\pm0.07}$ |
| BO | $0.80_{\pm0.10}$ | $0.82_{\pm0.01}$ | $1.64_{\pm0.02}$ | $1.09_{\pm0.01}$ | $1.23_{\pm0.02}$ | $0.80_{\pm0.10}$ | $0.83_{\pm0.01}$ | $1.65_{\pm0.02}$ | $1.10_{\pm0.01}$ | $0.99_{\pm0.01}$ | $0.77_{\pm0.09}$ | $0.82_{\pm0.01}$ | $1.65_{\pm0.02}$ | $1.10_{\pm0.01}$ | $0.83_{\pm0.01}$ |
| CBAS | $0.00_{\pm0.00}$ | $-43.12_{\pm0.18}$ | $-43.12_{\pm0.18}$ | $-43.12_{\pm0.18}$ | $-43.12_{\pm0.18}$ | $0.00_{\pm0.00}$ | $-43.35_{\pm0.18}$ | $-43.35_{\pm0.18}$ | $-43.35_{\pm0.18}$ | $-43.35_{\pm0.18}$ | $0.00_{\pm0.00}$ | $-43.42_{\pm0.18}$ | $-43.42_{\pm0.18}$ | $-43.42_{\pm0.18}$ | $-43.42_{\pm0.18}$ |
| CCDDEA | $0.73_{\pm0.07}$ | $0.73_{\pm0.09}$ | $1.45_{\pm0.19}$ | $0.97_{\pm0.12}$ | $1.09_{\pm0.14}$ | $0.73_{\pm0.08}$ | $0.77_{\pm0.07}$ | $1.55_{\pm0.14}$ | $1.03_{\pm0.09}$ | $0.79_{\pm0.07}$ | - | | | | |
| CMAES | $0.96_{\pm0.02}$ | $0.98_{\pm0.01}$ | $1.95_{\pm0.03}$ | $1.30_{\pm0.02}$ | $1.47_{\pm0.02}$ | - | | | | | - | | | | |
| TTDDEA | $0.24_{\pm0.09}$ | $0.24_{\pm0.09}$ | $0.49_{\pm0.18}$ | $0.33_{\pm0.13}$ | $0.37_{\pm0.13}$ | $0.23_{\pm0.07}$ | $0.17_{\pm0.12}$ | $0.34_{\pm0.23}$ | $0.23_{\pm0.15}$ | $0.21_{\pm0.14}$ | $0.23_{\pm0.07}$ | $0.15_{\pm0.13}$ | $0.29_{\pm0.26}$ | $0.20_{\pm0.17}$ | $0.15_{\pm0.13}$ |
| Trimentoring | $0.92_{\pm0.04}$ | $0.92_{\pm0.04}$ | $1.85_{\pm0.03}$ | $1.23_{\pm0.02}$ | $1.39_{\pm0.02}$ | $0.90_{\pm0.02}$ | $0.93_{\pm0.01}$ | $1.86_{\pm0.03}$ | $1.24_{\pm0.02}$ | $1.12_{\pm0.02}$ | $0.90_{\pm0.02}$ | $0.93_{\pm0.01}$ | $1.86_{\pm0.03}$ | $1.24_{\pm0.02}$ | $0.93_{\pm0.01}$ |
| Task $f(x^*_{OFF})$ | Mujoco 2 $0.03_{\pm0.01}$ | | | | | | | | | | | | | | |
| ARCOO | $0.03_{\pm0.01}$ | $1.00_{\pm0.00}$ | $1.00_{\pm0.00}$ | $1.00_{\pm0.00}$ | $1.00_{\pm0.00}$ | $0.42_{\pm0.13}$ | $0.67_{\pm0.10}$ | $1.13_{\pm0.08}$ | $0.83_{\pm0.07}$ | $0.77_{\pm0.08}$ | $0.51_{\pm0.08}$ | $0.68_{\pm0.03}$ | $1.37_{\pm0.07}$ | $0.91_{\pm0.04}$ | $0.69_{\pm0.03}$ |
| BO | $0.56_{\pm0.14}$ | $0.73_{\pm0.07}$ | $1.33_{\pm0.10}$ | $0.94_{\pm0.06}$ | $1.04_{\pm0.05}$ | $0.57_{\pm0.17}$ | $0.70_{\pm0.07}$ | $1.33_{\pm0.09}$ | $0.91_{\pm0.06}$ | $0.83_{\pm0.06}$ | $0.42_{\pm0.18}$ | $0.67_{\pm0.04}$ | $1.33_{\pm0.09}$ | $0.89_{\pm0.06}$ | $0.67_{\pm0.04}$ |
| CBAS | $-1.86_{\pm0.00}$ | $-185.68_{\pm0.45}$ | $-185.68_{\pm0.45}$ | $-185.68_{\pm0.45}$ | $-185.68_{\pm0.45}$ | $-1.86_{\pm0.00}$ | $-186.65_{\pm0.46}$ | $-186.65_{\pm0.46}$ | $-186.65_{\pm0.46}$ | $-186.65_{\pm0.46}$ | $-1.86_{\pm0.00}$ | $-186.97_{\pm0.46}$ | $-186.97_{\pm0.46}$ | $-186.97_{\pm0.46}$ | $-186.97_{\pm0.46}$ |
| CCDDEA | $0.13_{\pm0.29}$ | $0.75_{\pm0.05}$ | $1.49_{\pm0.10}$ | $1.00_{\pm0.07}$ | $1.12_{\pm0.08}$ | $0.34_{\pm0.14}$ | $0.74_{\pm0.05}$ | $1.48_{\pm0.10}$ | $0.99_{\pm0.06}$ | $0.89_{\pm0.06}$ | $0.34_{\pm0.14}$ | $0.74_{\pm0.07}$ | $1.49_{\pm0.14}$ | $0.99_{\pm0.09}$ | $0.74_{\pm0.07}$ |
| CMAES | $0.63_{\pm0.07}$ | $0.84_{\pm0.06}$ | $1.67_{\pm0.12}$ | $1.11_{\pm0.08}$ | $1.26_{\pm0.09}$ | - | | | | | - | | | | |
| TTDDEA | $-1.21_{\pm0.84}$ | $0.33_{\pm0.15}$ | $0.66_{\pm0.29}$ | $0.44_{\pm0.20}$ | $0.50_{\pm0.22}$ | $-0.62_{\pm0.80}$ | $0.27_{\pm0.20}$ | $0.53_{\pm0.39}$ | $0.35_{\pm0.26}$ | $0.32_{\pm0.24}$ | $-1.01_{\pm0.85}$ | $0.25_{\pm0.20}$ | $0.50_{\pm0.40}$ | $0.34_{\pm0.27}$ | $0.25_{\pm0.20}$ |
| Trimentoring | $0.49_{\pm0.10}$ | $0.82_{\pm0.07}$ | $1.63_{\pm0.13}$ | $1.09_{\pm0.09}$ | $1.23_{\pm0.10}$ | $0.43_{\pm0.13}$ | $0.81_{\pm0.07}$ | $1.63_{\pm0.14}$ | $1.08_{\pm0.10}$ | $0.98_{\pm0.09}$ | $0.41_{\pm0.12}$ | $0.80_{\pm0.06}$ | $1.59_{\pm0.13}$ | $1.06_{\pm0.09}$ | $0.80_{\pm0.06}$ |

Table 38: Overall results for mujoco unconstrained tasks with 128 solutions and 100th percentile evaluations. In this case, 0% of the values are missing near the worst value and another 20% near the optimal value. Details are the same as Table 5.

| Steps | t = 50 | | | | | t = 100 | | | | | t = 150 | | | | |
|---|---|---|---|---|---|---|---|---|---|---|---|---|---|---|---|
| Task $f(x^*_{OFF})$ | Mujoco 1 $0.50_{\pm0.00}$ | | | | | | | | | | | | | | |
| Metric | FS | SI | OI | SO | SO$_\omega$ | FS | SI | OI | SO | SO$_\omega$ | FS | SI | OI | SO | SO$_\omega$ |
| ARCOO | $0.50_{\pm0.00}$ | $1.00_{\pm0.01}$ | $1.00_{\pm0.00}$ | $1.00_{\pm0.00}$ | $1.00_{\pm0.00}$ | $0.64_{\pm0.06}$ | $0.72_{\pm0.10}$ | $1.10_{\pm0.07}$ | $0.86_{\pm0.07}$ | $0.81_{\pm0.08}$ | $0.75_{\pm0.08}$ | $0.67_{\pm0.03}$ | $1.33_{\pm0.08}$ | $0.89_{\pm0.04}$ | $0.67_{\pm0.03}$ |
| BO | $0.82_{\pm0.06}$ | $0.82_{\pm0.03}$ | $1.61_{\pm0.04}$ | $1.09_{\pm0.03}$ | $1.22_{\pm0.03}$ | $0.77_{\pm0.10}$ | $0.80_{\pm0.02}$ | $1.60_{\pm0.03}$ | $1.07_{\pm0.02}$ | $0.96_{\pm0.02}$ | $0.82_{\pm0.07}$ | $0.80_{\pm0.01}$ | $1.60_{\pm0.02}$ | $1.07_{\pm0.01}$ | $0.81_{\pm0.01}$ |
| CBAS | $0.00_{\pm0.00}$ | $-48.64_{\pm0.34}$ | $-48.64_{\pm0.34}$ | $-48.64_{\pm0.34}$ | $-48.64_{\pm0.34}$ | $0.00_{\pm0.00}$ | $-48.89_{\pm0.34}$ | $-48.89_{\pm0.34}$ | $-48.89_{\pm0.34}$ | $-48.89_{\pm0.34}$ | $0.00_{\pm0.00}$ | $-48.98_{\pm0.34}$ | $-48.98_{\pm0.34}$ | $-48.98_{\pm0.34}$ | $-48.98_{\pm0.34}$ |
| CCDDEA | $0.75_{\pm0.11}$ | $0.73_{\pm0.07}$ | $1.46_{\pm0.14}$ | $0.97_{\pm0.09}$ | $1.10_{\pm0.10}$ | $0.75_{\pm0.11}$ | $0.75_{\pm0.08}$ | $1.50_{\pm0.15}$ | $1.00_{\pm0.10}$ | $0.91_{\pm0.09}$ | $0.75_{\pm0.11}$ | $0.76_{\pm0.09}$ | $1.52_{\pm0.18}$ | $1.01_{\pm0.12}$ | $0.76_{\pm0.09}$ |
| CMAES | $0.97_{\pm0.01}$ | $0.98_{\pm0.00}$ | $1.97_{\pm0.01}$ | $1.31_{\pm0.01}$ | $1.48_{\pm0.01}$ | - | | | | | - | | | | |
| TTDDEA | $0.27_{\pm0.11}$ | $0.12_{\pm0.26}$ | $0.23_{\pm0.52}$ | $0.15_{\pm0.34}$ | $0.17_{\pm0.39}$ | $0.27_{\pm0.11}$ | $-0.02_{\pm0.31}$ | $-0.04_{\pm0.62}$ | $-0.03_{\pm0.41}$ | $-0.03_{\pm0.37}$ | $0.27_{\pm0.11}$ | $-0.07_{\pm0.33}$ | $-0.14_{\pm0.67}$ | $-0.10_{\pm0.44}$ | $-0.07_{\pm0.33}$ |
| Trimentoring | $0.94_{\pm0.04}$ | $0.92_{\pm0.02}$ | $1.84_{\pm0.04}$ | $1.23_{\pm0.03}$ | $1.39_{\pm0.03}$ | $0.94_{\pm0.04}$ | $0.94_{\pm0.03}$ | $1.87_{\pm0.06}$ | $1.25_{\pm0.04}$ | $1.13_{\pm0.03}$ | $0.93_{\pm0.03}$ | $0.94_{\pm0.03}$ | $1.88_{\pm0.06}$ | $1.25_{\pm0.04}$ | $0.94_{\pm0.03}$ |
| Task $f(x^*_{OFF})$ | Mujoco 2 $0.11_{\pm0.00}$ | | | | | | | | | | | | | | |
| ARCOO | $0.11_{\pm0.00}$ | $1.00_{\pm0.00}$ | $1.00_{\pm0.00}$ | $1.00_{\pm0.00}$ | $1.00_{\pm0.00}$ | $0.39_{\pm0.14}$ | $0.68_{\pm0.07}$ | $1.13_{\pm0.08}$ | $0.85_{\pm0.06}$ | $0.79_{\pm0.06}$ | $0.55_{\pm0.18}$ | $0.67_{\pm0.03}$ | $1.34_{\pm0.06}$ | $0.89_{\pm0.04}$ | $0.67_{\pm0.03}$ |
| BO | $0.51_{\pm0.16}$ | $0.71_{\pm0.06}$ | $1.35_{\pm0.06}$ | $0.93_{\pm0.06}$ | $1.03_{\pm0.06}$ | $0.50_{\pm0.17}$ | $0.68_{\pm0.03}$ | $1.35_{\pm0.07}$ | $0.90_{\pm0.04}$ | $0.80_{\pm0.04}$ | $0.44_{\pm0.14}$ | $0.68_{\pm0.04}$ | $1.36_{\pm0.07}$ | $0.90_{\pm0.05}$ | $0.68_{\pm0.04}$ |
| CBAS | $-1.86_{\pm0.00}$ | $-194.21_{\pm0.35}$ | $-194.21_{\pm0.35}$ | $-194.21_{\pm0.35}$ | $-194.21_{\pm0.35}$ | $-1.86_{\pm0.00}$ | $-195.23_{\pm0.35}$ | $-195.23_{\pm0.35}$ | $-195.23_{\pm0.35}$ | $-195.23_{\pm0.35}$ | $-1.86_{\pm0.00}$ | $-195.56_{\pm0.35}$ | $-195.56_{\pm0.35}$ | $-195.56_{\pm0.35}$ | $-195.56_{\pm0.35}$ |
| CCDDEA | $0.12_{\pm0.17}$ | $0.74_{\pm0.05}$ | $1.41_{\pm0.11}$ | $0.97_{\pm0.07}$ | $1.08_{\pm0.08}$ | $0.38_{\pm0.09}$ | $0.72_{\pm0.04}$ | $1.43_{\pm0.08}$ | $0.95_{\pm0.05}$ | $0.86_{\pm0.05}$ | $0.38_{\pm0.09}$ | $0.73_{\pm0.05}$ | $1.47_{\pm0.10}$ | $0.98_{\pm0.07}$ | $0.73_{\pm0.05}$ |
| CMAES | $0.60_{\pm0.05}$ | $0.85_{\pm0.06}$ | $1.71_{\pm0.12}$ | $1.14_{\pm0.08}$ | $1.28_{\pm0.09}$ | - | | | | | - | | | | |
| TTDDEA | $-1.86_{\pm0.00}$ | $-0.08_{\pm0.23}$ | $-0.17_{\pm0.47}$ | $-0.11_{\pm0.31}$ | $-0.13_{\pm0.35}$ | $-1.86_{\pm0.00}$ | $-0.23_{\pm0.26}$ | $-0.46_{\pm0.52}$ | $-0.30_{\pm0.34}$ | $-0.27_{\pm0.31}$ | $-1.64_{\pm0.58}$ | $-0.28_{\pm0.28}$ | $-0.57_{\pm0.56}$ | $-0.38_{\pm0.37}$ | $-0.28_{\pm0.28}$ |
| Trimentoring | $0.53_{\pm0.08}$ | $0.85_{\pm0.04}$ | $1.68_{\pm0.06}$ | $1.13_{\pm0.05}$ | $1.27_{\pm0.05}$ | $0.52_{\pm0.10}$ | $0.83_{\pm0.04}$ | $1.65_{\pm0.08}$ | $1.10_{\pm0.06}$ | $1.00_{\pm0.05}$ | $0.51_{\pm0.10}$ | $0.83_{\pm0.04}$ | $1.65_{\pm0.09}$ | $1.10_{\pm0.06}$ | $0.83_{\pm0.04}$ |

Table 39: Overall results for mujoco unconstrained tasks with 128 solutions and 100th percentile evaluations. In this case, 0% of the values are missing near the worst value and another 10% near the optimal value. Details are the same as Table 5.

| Steps | t = 50 | | | | | t = 100 | | | | | t = 150 | | | | |
|---|---|---|---|---|---|---|---|---|---|---|---|---|---|---|---|
| Task $f(x^*_{OFF})$ | Mujoco 1 $0.59_{\pm0.01}$ | | | | | | | | | | | | | | |
| Metric | FS | SI | OI | SO | SO$_\omega$ | FS | SI | OI | SO | SO$_\omega$ | FS | SI | OI | SO | SO$_\omega$ |
| ARCOO | $0.59_{\pm0.01}$ | $1.00_{\pm0.00}$ | $1.00_{\pm0.00}$ | $1.00_{\pm0.00}$ | $1.00_{\pm0.00}$ | $0.78_{\pm0.10}$ | $0.67_{\pm0.09}$ | $1.12_{\pm0.08}$ | $0.84_{\pm0.08}$ | $0.77_{\pm0.08}$ | $0.87_{\pm0.08}$ | $0.67_{\pm0.05}$ | $1.35_{\pm0.10}$ | $0.90_{\pm0.07}$ | $0.68_{\pm0.05}$ |
| BO | $0.75_{\pm0.07}$ | $0.76_{\pm0.02}$ | $1.50_{\pm0.02}$ | $1.01_{\pm0.02}$ | $1.13_{\pm0.02}$ | $0.70_{\pm0.08}$ | $0.76_{\pm0.01}$ | $1.51_{\pm0.01}$ | $1.01_{\pm0.01}$ | $0.76_{\pm0.01}$ | $0.78_{\pm0.08}$ | $0.75_{\pm0.01}$ | $1.51_{\pm0.01}$ | $1.00_{\pm0.01}$ | $0.76_{\pm0.01}$ |
| CBAS | $0.00_{\pm0.00}$ | $-57.79_{\pm0.58}$ | $-57.79_{\pm0.58}$ | $-57.79_{\pm0.58}$ | $-57.79_{\pm0.58}$ | $0.00_{\pm0.00}$ | $-58.10_{\pm0.58}$ | $-58.10_{\pm0.58}$ | $-58.10_{\pm0.58}$ | $-58.10_{\pm0.58}$ | $0.00_{\pm0.00}$ | $-58.20_{\pm0.59}$ | $-58.20_{\pm0.59}$ | $-58.20_{\pm0.59}$ | $-58.20_{\pm0.59}$ |
| CCDDEA | $0.71_{\pm0.07}$ | $0.66_{\pm0.11}$ | $1.31_{\pm0.22}$ | $0.88_{\pm0.15}$ | $0.99_{\pm0.16}$ | $0.70_{\pm0.06}$ | $0.66_{\pm0.05}$ | $1.32_{\pm0.09}$ | $0.88_{\pm0.06}$ | $0.80_{\pm0.06}$ | $0.70_{\pm0.06}$ | $0.66_{\pm0.05}$ | $1.33_{\pm0.10}$ | $0.88_{\pm0.07}$ | $0.67_{\pm0.05}$ |
| CMAES | $0.97_{\pm0.01}$ | $0.98_{\pm0.01}$ | $1.96_{\pm0.01}$ | $1.31_{\pm0.01}$ | $1.48_{\pm0.01}$ | - | | | | | - | | | | |
| TTDDEA | $0.34_{\pm0.12}$ | $-1.41_{\pm1.54}$ | $-2.82_{\pm3.07}$ | $-1.88_{\pm2.05}$ | $-2.12_{\pm2.31}$ | $0.34_{\pm0.12}$ | $-1.90_{\pm1.99}$ | $-3.80_{\pm3.99}$ | $-2.53_{\pm2.66}$ | $-2.29_{\pm2.40}$ | $0.34_{\pm0.12}$ | $-2.05_{\pm2.15}$ | $-4.10_{\pm4.30}$ | $-2.74_{\pm2.87}$ | $-2.06_{\pm2.16}$ |
| Trimentoring | $0.93_{\pm0.04}$ | $0.92_{\pm0.04}$ | $1.84_{\pm0.04}$ | $1.23_{\pm0.02}$ | $1.38_{\pm0.03}$ | $0.94_{\pm0.04}$ | $0.94_{\pm0.02}$ | $1.87_{\pm0.06}$ | $1.25_{\pm0.03}$ | $1.12_{\pm0.03}$ | $0.93_{\pm0.04}$ | $0.94_{\pm0.03}$ | $1.88_{\pm0.05}$ | $1.25_{\pm0.04}$ | $0.94_{\pm0.03}$ |
| Task $f(x^*_{OFF})$ | Mujoco 2 $0.23_{\pm0.01}$ | | | | | | | | | | | | | | |
| ARCOO | $0.23_{\pm0.01}$ | $1.00_{\pm0.00}$ | $1.00_{\pm0.00}$ | $1.00_{\pm0.00}$ | $1.00_{\pm0.00}$ | $0.45_{\pm0.12}$ | $0.67_{\pm0.10}$ | $1.12_{\pm0.10}$ | $0.83_{\pm0.10}$ | $0.77_{\pm0.08}$ | $0.53_{\pm0.15}$ | $0.67_{\pm0.07}$ | $1.33_{\pm0.14}$ | $0.89_{\pm0.06}$ | $0.67_{\pm0.07}$ |
| BO | $0.43_{\pm0.19}$ | $0.70_{\pm0.04}$ | $1.24_{\pm0.06}$ | $0.90_{\pm0.04}$ | $0.99_{\pm0.04}$ | $0.55_{\pm0.28}$ | $0.67_{\pm0.04}$ | $1.25_{\pm0.06}$ | $0.87_{\pm0.04}$ | $0.80_{\pm0.04}$ | $0.46_{\pm0.17}$ | $0.63_{\pm0.03}$ | $1.26_{\pm0.06}$ | $0.84_{\pm0.04}$ | $0.63_{\pm0.03}$ |
| CBAS | $-1.86_{\pm0.00}$ | $-205.86_{\pm0.46}$ | $-205.86_{\pm0.46}$ | $-205.86_{\pm0.46}$ | $-205.86_{\pm0.46}$ | $-1.86_{\pm0.00}$ | $-206.94_{\pm0.46}$ | $-206.94_{\pm0.46}$ | $-206.94_{\pm0.46}$ | $-206.94_{\pm0.46}$ | $-1.86_{\pm0.00}$ | $-207.29_{\pm0.46}$ | $-207.29_{\pm0.46}$ | $-207.29_{\pm0.46}$ | $-207.29_{\pm0.46}$ |
| CCDDEA | $0.00_{\pm0.23}$ | $0.64_{\pm0.08}$ | $1.28_{\pm0.15}$ | $0.86_{\pm0.10}$ | $0.96_{\pm0.11}$ | $0.38_{\pm0.11}$ | $0.64_{\pm0.05}$ | $1.28_{\pm0.10}$ | $0.86_{\pm0.07}$ | $0.77_{\pm0.06}$ | $0.38_{\pm0.11}$ | $0.66_{\pm0.07}$ | $1.32_{\pm0.14}$ | $0.88_{\pm0.10}$ | $0.66_{\pm0.07}$ |
| CMAES | $0.69_{\pm0.07}$ | $0.84_{\pm0.07}$ | $1.67_{\pm0.15}$ | $1.11_{\pm0.10}$ | $1.26_{\pm0.11}$ | - | | | | | - | | | | |
| TTDDEA | $-0.69_{\pm0.91}$ | $0.07_{\pm0.40}$ | $0.13_{\pm0.80}$ | $0.09_{\pm0.53}$ | $0.10_{\pm0.60}$ | $-0.49_{\pm0.80}$ | $-0.14_{\pm0.60}$ | $-0.29_{\pm1.21}$ | $-0.19_{\pm0.81}$ | $-0.17_{\pm0.73}$ | $-0.44_{\pm0.83}$ | $-0.18_{\pm0.63}$ | $-0.35_{\pm1.27}$ | $-0.24_{\pm0.84}$ | $-0.18_{\pm0.64}$ |
| Trimentoring | $0.59_{\pm0.06}$ | $0.85_{\pm0.04}$ | $1.68_{\pm0.07}$ | $1.12_{\pm0.05}$ | $1.27_{\pm0.06}$ | $0.58_{\pm0.07}$ | $0.84_{\pm0.04}$ | $1.68_{\pm0.08}$ | $1.12_{\pm0.05}$ | $1.01_{\pm0.05}$ | $0.56_{\pm0.07}$ | $0.83_{\pm0.04}$ | $1.67_{\pm0.07}$ | $1.11_{\pm0.05}$ | $0.84_{\pm0.04}$ |

Table 40: Overall results for mujoco unconstrained tasks with 128 solutions and 100th percentile evaluations. In this case, 10% of the values are missing near the worst value and another 40% near the optimal value. Details are the same as Table 5.

| Steps | t = 50 | | | | | t = 100 | | | | | t = 150 | | | | |
|---|---|---|---|---|---|---|---|---|---|---|---|---|---|---|---|
| Task $f(x^*_{OFF})$ | Mujoco 1 $0.40_{\pm0.00}$ | | | | | | | | | | | | | | |
| Metric | FS | SI | OI | SO | SO$_\omega$ | FS | SI | OI | SO | SO$_\omega$ | FS | SI | OI | SO | SO$_\omega$ |
| ARCOO | $0.45_{\pm0.14}$ | $0.94_{\pm0.15}$ | $1.01_{\pm0.03}$ | $0.97_{\pm0.09}$ | $0.98_{\pm0.06}$ | $0.77_{\pm0.11}$ | $0.69_{\pm0.04}$ | $1.27_{\pm0.13}$ | $0.89_{\pm0.06}$ | $0.82_{\pm0.05}$ | $0.84_{\pm0.10}$ | $0.74_{\pm0.05}$ | $1.47_{\pm0.11}$ | $0.98_{\pm0.07}$ | $0.74_{\pm0.06}$ |
| BO | $0.81_{\pm0.08}$ | $0.84_{\pm0.02}$ | $1.66_{\pm0.02}$ | $1.11_{\pm0.02}$ | $1.25_{\pm0.02}$ | $0.82_{\pm0.07}$ | $0.83_{\pm0.01}$ | $1.66_{\pm0.02}$ | $1.11_{\pm0.01}$ | $1.00_{\pm0.01}$ | $0.86_{\pm0.06}$ | $0.83_{\pm0.01}$ | $1.66_{\pm0.01}$ | $1.11_{\pm0.01}$ | $0.83_{\pm0.01}$ |
| CBAS | $0.00_{\pm0.00}$ | $-38.87_{\pm0.22}$ | $-38.87_{\pm0.22}$ | $-38.87_{\pm0.22}$ | $-38.87_{\pm0.22}$ | $0.00_{\pm0.00}$ | $-39.08_{\pm0.22}$ | $-39.08_{\pm0.22}$ | $-39.08_{\pm0.22}$ | $-39.08_{\pm0.22}$ | $0.00_{\pm0.00}$ | $-39.15_{\pm0.22}$ | $-39.15_{\pm0.22}$ | $-39.15_{\pm0.22}$ | $-39.15_{\pm0.22}$ |
| CCDDEA | $0.71_{\pm0.20}$ | $0.76_{\pm0.05}$ | $1.51_{\pm0.10}$ | $1.01_{\pm0.07}$ | $1.14_{\pm0.08}$ | $0.74_{\pm0.10}$ | $0.79_{\pm0.06}$ | $1.57_{\pm0.13}$ | $1.05_{\pm0.08}$ | $0.95_{\pm0.08}$ | $0.74_{\pm0.10}$ | $0.80_{\pm0.07}$ | $1.59_{\pm0.14}$ | $1.06_{\pm0.10}$ | $0.80_{\pm0.07}$ |
| CMAES | $0.97_{\pm0.01}$ | $0.98_{\pm0.01}$ | $1.96_{\pm0.01}$ | $1.31_{\pm0.01}$ | $1.48_{\pm0.01}$ | - | | | | | - | | | | |
| TTDDEA | $0.22_{\pm0.11}$ | $0.36_{\pm0.12}$ | $0.73_{\pm0.25}$ | $0.48_{\pm0.17}$ | $0.55_{\pm0.19}$ | $0.21_{\pm0.10}$ | $0.30_{\pm0.16}$ | $0.61_{\pm0.32}$ | $0.41_{\pm0.21}$ | $0.36_{\pm0.19}$ | $0.24_{\pm0.07}$ | $0.28_{\pm0.17}$ | $0.57_{\pm0.34}$ | $0.38_{\pm0.23}$ | $0.28_{\pm0.17}$ |
| Trimentoring | $0.89_{\pm0.04}$ | $0.91_{\pm0.02}$ | $1.81_{\pm0.04}$ | $1.21_{\pm0.03}$ | $1.36_{\pm0.03}$ | $0.87_{\pm0.07}$ | $0.92_{\pm0.04}$ | $1.84_{\pm0.08}$ | $1.23_{\pm0.05}$ | $1.11_{\pm0.05}$ | $0.86_{\pm0.06}$ | $0.92_{\pm0.04}$ | $1.83_{\pm0.09}$ | $1.22_{\pm0.06}$ | $0.92_{\pm0.04}$ |
| Task $f(x^*_{OFF})$ | Mujoco 2 $-0.04_{\pm0.00}$ | | | | | | | | | | | | | | |
| ARCOO | $-0.03_{\pm0.02}$ | $0.98_{\pm0.04}$ | $1.00_{\pm0.00}$ | $0.99_{\pm0.02}$ | $0.99_{\pm0.01}$ | $0.38_{\pm0.09}$ | $0.68_{\pm0.11}$ | $1.17_{\pm0.10}$ | $0.85_{\pm0.09}$ | $0.79_{\pm0.10}$ | $0.57_{\pm0.39}$ | $0.69_{\pm0.05}$ | $1.39_{\pm0.10}$ | $0.93_{\pm0.07}$ | $0.70_{\pm0.05}$ |
| BO | $0.48_{\pm0.12}$ | $0.75_{\pm0.04}$ | $1.43_{\pm0.07}$ | $0.99_{\pm0.04}$ | $1.10_{\pm0.05}$ | $0.42_{\pm0.10}$ | $0.73_{\pm0.04}$ | $1.44_{\pm0.07}$ | $0.97_{\pm0.04}$ | $0.88_{\pm0.04}$ | $0.50_{\pm0.18}$ | $0.72_{\pm0.03}$ | $1.45_{\pm0.07}$ | $0.96_{\pm0.04}$ | $0.72_{\pm0.03}$ |
| CBAS | $-1.86_{\pm0.00}$ | $-179.29_{\pm0.34}$ | $-179.29_{\pm0.34}$ | $-179.29_{\pm0.34}$ | $-179.29_{\pm0.34}$ | $-1.86_{\pm0.00}$ | $-180.23_{\pm0.34}$ | $-180.23_{\pm0.34}$ | $-180.23_{\pm0.34}$ | $-180.23_{\pm0.34}$ | $-1.86_{\pm0.00}$ | $-180.54_{\pm0.34}$ | $-180.54_{\pm0.34}$ | $-180.54_{\pm0.34}$ | $-180.54_{\pm0.34}$ |
| CCDDEA | $0.26_{\pm0.17}$ | $0.80_{\pm0.05}$ | $1.60_{\pm0.10}$ | $1.06_{\pm0.07}$ | $1.20_{\pm0.08}$ | $0.26_{\pm0.13}$ | $0.77_{\pm0.06}$ | $1.53_{\pm0.12}$ | $1.02_{\pm0.08}$ | $0.92_{\pm0.07}$ | $0.26_{\pm0.13}$ | $0.75_{\pm0.07}$ | $1.50_{\pm0.14}$ | $1.00_{\pm0.10}$ | $0.76_{\pm0.07}$ |
| CMAES | $0.66_{\pm0.06}$ | $0.87_{\pm0.05}$ | $1.74_{\pm0.11}$ | $1.16_{\pm0.07}$ | $1.31_{\pm0.08}$ | - | | | | | - | | | | |
| TTDDEA | $-0.29_{\pm0.36}$ | $0.49_{\pm0.14}$ | $0.98_{\pm0.28}$ | $0.65_{\pm0.19}$ | $0.73_{\pm0.21}$ | $-0.26_{\pm0.34}$ | $0.42_{\pm0.17}$ | $0.84_{\pm0.33}$ | $0.56_{\pm0.22}$ | $0.51_{\pm0.20}$ | $-0.26_{\pm0.39}$ | $0.41_{\pm0.17}$ | $0.81_{\pm0.35}$ | $0.54_{\pm0.23}$ | $0.41_{\pm0.17}$ |
| Trimentoring | $0.66_{\pm0.23}$ | $0.85_{\pm0.04}$ | $1.69_{\pm0.08}$ | $1.13_{\pm0.06}$ | $1.27_{\pm0.06}$ | $0.53_{\pm0.12}$ | $0.86_{\pm0.05}$ | $1.72_{\pm0.10}$ | $1.15_{\pm0.07}$ | $1.03_{\pm0.06}$ | $0.51_{\pm0.13}$ | $0.86_{\pm0.06}$ | $1.71_{\pm0.11}$ | $1.14_{\pm0.07}$ | $0.86_{\pm0.06}$ |

Table 41: Overall results for mujoco unconstrained tasks with 128 solutions and 100th percentile evaluations. In this case, 20% of the values are missing near the worst value and another 30% near the optimal value. Details are the same as Table 5.

| Steps | t = 50 | | | | | t = 100 | | | | | t = 150 | | | | |
|---|---|---|---|---|---|---|---|---|---|---|---|---|---|---|---|
| **Task** $f(x^*_{OFF})$ | Mujoco 1 $0.45_{\pm0.00}$ | | | | | | | | | | | | | | |
| Metric | FS | SI | OI | SO | SO$_\omega$ | FS | SI | OI | SO | SO$_\omega$ | FS | SI | OI | SO | SO$_\omega$ |
| ARCOO | 0.50±0.10 | 0.91±0.14 | 1.00±0.01 | 0.95±0.09 | 0.96±0.06 | 0.80±0.10 | 0.70±0.08 | 1.29±0.08 | 0.91±0.07 | 0.83±0.07 | 0.80±0.08 | 0.74±0.02 | 1.48±0.05 | 0.99±0.03 | 0.74±0.02 |
| BO | 0.84±0.08 | 0.82±0.01 | 1.63±0.02 | 1.09±0.02 | 1.23±0.02 | 0.76±0.10 | 0.82±0.01 | 1.63±0.01 | 1.09±0.01 | 0.98±0.01 | 0.75±0.08 | 0.82±0.01 | 1.63±0.01 | 1.09±0.01 | 0.82±0.01 |
| CBAS | 0.00±0.00 | -43.12±0.18 | -43.12±0.18 | -43.12±0.18 | -43.12±0.18 | 0.00±0.00 | -43.35±0.18 | -43.35±0.18 | -43.35±0.18 | -43.35±0.18 | 0.00±0.00 | -43.42±0.18 | -43.42±0.18 | -43.42±0.18 | -43.42±0.18 |
| CCDDEA | 0.65±0.14 | 0.76±0.06 | 1.52±0.11 | 1.01±0.08 | 1.14±0.09 | 0.65±0.14 | 0.74±0.09 | 1.48±0.18 | 0.99±0.12 | 0.89±0.11 | 0.65±0.14 | 0.74±0.11 | 1.47±0.22 | 0.98±0.14 | 0.74±0.11 |
| CMAES | 0.97±0.01 | 0.98±0.01 | 1.96±0.02 | 1.31±0.01 | 1.47±0.02 | - | - | - | - | - | - | - | - | - | - |
| TTDDEA | 0.30±0.07 | 0.35±0.09 | 0.69±0.18 | 0.46±0.12 | 0.52±0.14 | 0.28±0.10 | 0.30±0.10 | 0.59±0.20 | 0.40±0.13 | 0.36±0.12 | 0.30±0.07 | 0.28±0.10 | 0.56±0.20 | 0.38±0.14 | 0.28±0.10 |
| Trimentoring | 0.91±0.09 | 0.91±0.04 | 1.83±0.09 | 1.22±0.06 | 1.37±0.07 | 0.86±0.14 | 0.91±0.06 | 1.82±0.12 | 1.22±0.08 | 1.10±0.07 | 0.85±0.13 | 0.90±0.08 | 1.81±0.16 | 1.21±0.11 | 0.91±0.08 |
| **Task** $f(x^*_{OFF})$ | Mujoco 2 $0.03_{\pm0.01}$ | | | | | | | | | | | | | | |
| ARCOO | 0.05±0.06 | 0.97±0.08 | 1.00±0.00 | 0.98±0.05 | 0.99±0.03 | 0.40±0.15 | 0.71±0.07 | 1.21±0.06 | 0.89±0.05 | 0.82±0.06 | 0.50±0.12 | 0.72±0.02 | 1.44±0.05 | 0.96±0.03 | 0.72±0.02 |
| BO | 0.45±0.12 | 0.75±0.04 | 1.43±0.10 | 0.98±0.05 | 1.10±0.06 | 0.57±0.17 | 0.73±0.04 | 1.43±0.09 | 0.97±0.05 | 0.88±0.04 | 0.48±0.14 | 0.71±0.05 | 1.43±0.10 | 0.95±0.06 | 0.72±0.05 |
| CBAS | -1.86±0.00 | -185.68±0.45 | -185.68±0.45 | -185.68±0.45 | -185.68±0.45 | -1.86±0.00 | -186.65±0.46 | -186.65±0.46 | -186.65±0.46 | -186.65±0.46 | -1.86±0.00 | -186.97±0.46 | -186.97±0.46 | -186.97±0.46 | -186.97±0.46 |
| CCDDEA | 0.19±0.25 | 0.73±0.05 | 1.44±0.09 | 0.97±0.06 | 1.09±0.06 | 0.26±0.14 | 0.70±0.04 | 1.39±0.07 | 0.93±0.05 | 0.84±0.04 | 0.26±0.14 | 0.69±0.05 | 1.37±0.10 | 0.91±0.07 | 0.69±0.05 |
| CMAES | 0.68±0.14 | 0.84±0.06 | 1.68±0.13 | 1.12±0.08 | 1.26±0.10 | - | - | - | - | - | - | - | - | - | - |
| TTDDEA | -0.26±0.14 | 0.24±0.11 | 0.48±0.23 | 0.32±0.15 | 0.36±0.17 | -0.26±0.14 | 0.18±0.13 | 0.36±0.25 | 0.24±0.17 | 0.22±0.15 | -0.28±0.14 | 0.16±0.14 | 0.32±0.27 | 0.21±0.18 | 0.16±0.14 |
| Trimentoring | 0.64±0.08 | 0.84±0.03 | 1.69±0.06 | 1.12±0.04 | 1.27±0.05 | 0.59±0.08 | 0.83±0.04 | 1.67±0.08 | 1.11±0.05 | 1.00±0.05 | 0.59±0.09 | 0.83±0.04 | 1.66±0.09 | 1.11±0.06 | 0.83±0.04 |

Table 42: Overall results for mujoco unconstrained tasks with 128 solutions and 100th percentile evaluations. In this case, 25% of the values are missing near the worst value and another 25% near the optimal value. Details are the same as Table 5.

| Steps | t = 50 | | | | | t = 100 | | | | | t = 150 | | | | |
|---|---|---|---|---|---|---|---|---|---|---|---|---|---|---|---|
| **Task** $f(x^*_{OFF})$ | Mujoco 1 $0.47_{\pm0.00}$ | | | | | | | | | | | | | | |
| Metric | FS | SI | OI | SO | SO$_\omega$ | FS | SI | OI | SO | SO$_\omega$ | FS | SI | OI | SO | SO$_\omega$ |
| ARCOO | 0.50±0.04 | 0.94±0.08 | 1.00±0.01 | 0.97±0.04 | 0.98±0.03 | 0.77±0.08 | 0.68±0.05 | 1.26±0.06 | 0.89±0.04 | 0.81±0.04 | 0.82±0.07 | 0.73±0.02 | 1.47±0.05 | 0.98±0.03 | 0.74±0.02 |
| BO | 0.85±0.08 | 0.82±0.02 | 1.61±0.02 | 1.08±0.02 | 1.22±0.02 | 0.80±0.08 | 0.81±0.01 | 1.62±0.01 | 1.08±0.01 | 0.98±0.01 | 0.82±0.08 | 0.81±0.01 | 1.62±0.01 | 1.08±0.01 | 0.81±0.01 |
| CBAS | 0.00±0.00 | -45.62±0.25 | -45.62±0.25 | -45.62±0.25 | -45.62±0.25 | 0.00±0.00 | -45.86±0.26 | -45.86±0.26 | -45.86±0.26 | -45.86±0.26 | 0.00±0.00 | -45.94±0.26 | -45.94±0.26 | -45.94±0.26 | -45.94±0.26 |
| CCDDEA | 0.69±0.17 | 0.72±0.08 | 1.45±0.16 | 0.96±0.11 | 1.08±0.12 | 0.73±0.13 | 0.77±0.11 | 1.53±0.21 | 1.02±0.14 | 0.92±0.13 | 0.73±0.13 | 0.78±0.14 | 1.56±0.24 | 1.04±0.16 | 0.78±0.12 |
| CMAES | 0.97±0.02 | 0.98±0.01 | 1.96±0.03 | 1.30±0.02 | 1.47±0.02 | - | - | - | - | - | - | - | - | - | - |
| TTDDEA | 0.30±0.06 | 0.26±0.12 | 0.53±0.24 | 0.35±0.16 | 0.40±0.18 | 0.30±0.08 | 0.18±0.15 | 0.37±0.31 | 0.25±0.21 | 0.22±0.19 | 0.32±0.09 | 0.16±0.17 | 0.33±0.34 | 0.22±0.22 | 0.16±0.17 |
| Trimentoring | 0.94±0.04 | 0.93±0.02 | 1.85±0.03 | 1.24±0.02 | 1.39±0.02 | 0.91±0.04 | 0.94±0.02 | 1.88±0.04 | 1.25±0.03 | 1.13±0.03 | 0.91±0.05 | 0.94±0.02 | 1.89±0.05 | 1.26±0.03 | 0.95±0.02 |
| **Task** $f(x^*_{OFF})$ | Mujoco 2 $0.07_{\pm0.00}$ | | | | | | | | | | | | | | |
| ARCOO | 0.07±0.00 | 1.00±0.00 | 1.00±0.00 | 1.00±0.00 | 1.00±0.00 | 0.42±0.12 | 0.66±0.06 | 1.19±0.09 | 0.84±0.07 | 0.77±0.06 | 0.46±0.14 | 0.70±0.03 | 1.40±0.06 | 0.94±0.04 | 0.70±0.03 |
| BO | 0.41±0.11 | 0.70±0.07 | 1.32±0.10 | 0.92±0.07 | 1.02±0.08 | 0.47±0.21 | 0.71±0.07 | 1.33±0.10 | 0.93±0.08 | 0.84±0.07 | 0.58±0.26 | 0.67±0.05 | 1.34±0.10 | 0.89±0.07 | 0.67±0.05 |
| CBAS | -1.86±0.00 | -189.94±0.32 | -189.94±0.32 | -189.94±0.32 | -189.94±0.32 | -1.86±0.00 | -190.93±0.33 | -190.93±0.33 | -190.93±0.33 | -190.93±0.33 | -1.86±0.00 | -191.26±0.33 | -191.26±0.33 | -191.26±0.33 | -191.26±0.33 |
| CCDDEA | 0.08±0.25 | 0.71±0.06 | 1.42±0.12 | 0.95±0.08 | 1.07±0.09 | 0.33±0.18 | 0.69±0.09 | 1.39±0.19 | 0.93±0.12 | 0.84±0.11 | 0.33±0.18 | 0.70±0.11 | 1.40±0.23 | 0.93±0.15 | 0.70±0.11 |
| CMAES | 0.79±0.15 | 0.83±0.04 | 1.66±0.08 | 1.11±0.05 | 1.25±0.06 | - | - | - | - | - | - | - | - | - | - |
| TTDDEA | -0.18±0.12 | 0.24±0.13 | 0.47±0.25 | 0.32±0.17 | 0.36±0.19 | -0.21±0.14 | 0.18±0.15 | 0.35±0.30 | 0.24±0.20 | 0.21±0.18 | -0.22±0.11 | 0.15±0.16 | 0.31±0.32 | 0.21±0.21 | 0.16±0.16 |
| Trimentoring | 0.65±0.10 | 0.83±0.04 | 1.61±0.12 | 1.10±0.04 | 1.23±0.06 | 0.56±0.21 | 0.81±0.07 | 1.56±0.14 | 1.07±0.08 | 0.97±0.07 | 0.58±0.12 | 0.78±0.08 | 1.55±0.16 | 1.03±0.10 | 0.78±0.08 |

Table 43: Overall results for mujoco unconstrained tasks with 128 solutions and 100th percentile evaluations. In this case, 30% of the values are missing near the worst value and another 20% near the optimal value. Details are the same as Table 5.

| Steps | t = 50 | | | | | t = 100 | | | | | t = 150 | | | | |
|---|---|---|---|---|---|---|---|---|---|---|---|---|---|---|---|
| **Task** $f(x^*_{OFF})$ | Mujoco 1 $0.50_{\pm0.00}$ | | | | | | | | | | | | | | |
| Metric | FS | SI | OI | SO | SO$_\omega$ | FS | SI | OI | SO | SO$_\omega$ | FS | SI | OI | SO | SO$_\omega$ |
| ARCOO | 0.54±0.07 | 0.95±0.10 | 1.01±0.03 | 0.97±0.04 | 0.99±0.02 | 0.78±0.08 | 0.69±0.05 | 1.24±0.06 | 0.88±0.05 | 0.81±0.05 | 0.84±0.05 | 0.73±0.04 | 1.47±0.08 | 0.98±0.05 | 0.73±0.04 |
| BO | 0.80±0.10 | 0.81±0.02 | 1.59±0.02 | 1.07±0.02 | 1.20±0.02 | 0.79±0.11 | 0.80±0.01 | 1.60±0.01 | 1.07±0.01 | 0.98±0.01 | 0.77±0.10 | 0.80±0.01 | 1.60±0.01 | 1.07±0.01 | 0.81±0.01 |
| CBAS | 0.00±0.00 | -48.64±0.34 | -48.64±0.34 | -48.64±0.34 | -48.64±0.34 | 0.00±0.00 | -48.89±0.34 | -48.89±0.34 | -48.89±0.34 | -48.89±0.34 | 0.00±0.00 | -48.98±0.34 | -48.98±0.34 | -48.98±0.34 | -48.98±0.34 |
| CCDDEA | 0.73±0.14 | 0.72±0.08 | 1.44±0.15 | 0.96±0.10 | 1.08±0.11 | 0.73±0.14 | 0.76±0.12 | 1.53±0.24 | 1.02±0.16 | 0.92±0.14 | 0.73±0.14 | 0.78±0.14 | 1.56±0.28 | 1.04±0.18 | 0.78±0.14 |
| CMAES | 0.97±0.02 | 0.98±0.01 | 1.96±0.02 | 1.30±0.01 | 1.47±0.01 | - | - | - | - | - | - | - | - | - | - |
| TTDDEA | 0.35±0.08 | 0.12±0.13 | 0.25±0.26 | 0.17±0.17 | 0.19±0.20 | 0.32±0.08 | 0.06±0.16 | 0.12±0.33 | 0.08±0.22 | 0.07±0.20 | 0.33±0.08 | 0.04±0.17 | 0.08±0.34 | 0.06±0.22 | 0.04±0.17 |
| Trimentoring | 0.94±0.03 | 0.92±0.03 | 1.84±0.06 | 1.23±0.04 | 1.38±0.04 | 0.93±0.03 | 0.94±0.04 | 1.87±0.07 | 1.25±0.05 | 1.12±0.04 | 0.92±0.04 | 0.94±0.04 | 1.87±0.08 | 1.25±0.05 | 0.94±0.04 |
| **Task** $f(x^*_{OFF})$ | Mujoco 2 $0.11_{\pm0.00}$ | | | | | | | | | | | | | | |
| ARCOO | 0.11±0.00 | 1.00±0.00 | 1.00±0.00 | 1.00±0.00 | 1.00±0.00 | 0.41±0.17 | 0.71±0.12 | 1.14±0.07 | 0.87±0.06 | 0.81±0.09 | 0.62±0.33 | 0.67±0.05 | 1.35±0.09 | 0.90±0.06 | 0.68±0.05 |
| BO | 0.42±0.13 | 0.73±0.08 | 1.33±0.09 | 0.94±0.07 | 1.04±0.07 | 0.46±0.12 | 0.69±0.05 | 1.33±0.09 | 0.91±0.06 | 0.82±0.06 | 0.43±0.11 | 0.67±0.04 | 1.33±0.09 | 0.89±0.06 | 0.67±0.04 |
| CBAS | -1.86±0.00 | -194.21±0.35 | -194.21±0.35 | -194.21±0.35 | -194.21±0.35 | -1.86±0.00 | -195.23±0.35 | -195.23±0.35 | -195.23±0.35 | -195.23±0.35 | -1.86±0.00 | -195.56±0.35 | -195.56±0.35 | -195.56±0.35 | -195.56±0.35 |
| CCDDEA | 0.18±0.21 | 0.69±0.09 | 1.35±0.14 | 0.92±0.10 | 1.03±0.11 | 0.29±0.11 | 0.65±0.03 | 1.29±0.07 | 0.86±0.04 | 0.78±0.04 | 0.29±0.11 | 0.64±0.03 | 1.27±0.07 | 0.85±0.04 | 0.64±0.03 |
| CMAES | 0.76±0.17 | 0.84±0.05 | 1.68±0.09 | 1.12±0.06 | 1.26±0.07 | - | - | - | - | - | - | - | - | - | - |
| TTDDEA | -0.15±0.14 | 0.21±0.17 | 0.42±0.33 | 0.28±0.22 | 0.31±0.25 | -0.15±0.11 | 0.15±0.19 | 0.31±0.37 | 0.21±0.25 | 0.19±0.22 | -0.19±0.14 | 0.13±0.19 | 0.26±0.38 | 0.17±0.25 | 0.13±0.19 |
| Trimentoring | 0.75±0.14 | 0.81±0.07 | 1.62±0.12 | 1.08±0.08 | 1.22±0.09 | 0.64±0.13 | 0.80±0.06 | 1.58±0.11 | 1.06±0.08 | 0.96±0.07 | 0.65±0.11 | 0.79±0.06 | 1.58±0.12 | 1.05±0.08 | 0.79±0.06 |

Table 44: Overall results for mujoco unconstrained tasks with 128 solutions and 100th percentile evaluations. In this case, 40% of the values are missing near the worst value and another 10% near the optimal value. Details are the same as Table 5.

| Steps | t = 50 | | | | | t = 100 | | | | | t = 150 | | | | |
|---|---|---|---|---|---|---|---|---|---|---|---|---|---|---|---|
| **Task** $f(x^*_{OFF})$ | Mujoco 1 $0.59_{\pm0.01}$ | | | | | | | | | | | | | | |
| Metric | FS | SI | OI | SO | SO$_\omega$ | FS | SI | OI | SO | SO$_\omega$ | FS | SI | OI | SO | SO$_\omega$ |
| ARCOO | 0.62±0.05 | 0.95±0.09 | 1.00±0.01 | 0.98±0.05 | 0.98±0.04 | 0.89±0.06 | 0.68±0.04 | 1.27±0.09 | 0.89±0.05 | 0.81±0.04 | 0.92±0.04 | 0.75±0.03 | 1.49±0.08 | 1.00±0.04 | 0.75±0.03 |
| BO | 0.79±0.10 | 0.77±0.02 | 1.50±0.03 | 1.02±0.02 | 1.14±0.03 | 0.71±0.12 | 0.77±0.01 | 1.51±0.03 | 1.01±0.02 | 0.98±0.02 | 0.82±0.10 | 0.75±0.01 | 1.51±0.02 | 1.00±0.01 | 0.76±0.01 |
| CBAS | 0.00±0.00 | -57.79±0.58 | -57.79±0.58 | -57.79±0.58 | -57.79±0.58 | 0.00±0.00 | -58.10±0.58 | -58.10±0.58 | -58.10±0.58 | -58.10±0.58 | 0.00±0.00 | -58.20±0.59 | -58.20±0.59 | -58.20±0.59 | -58.20±0.59 |
| CCDDEA | 0.74±0.11 | 0.65±0.14 | 1.30±0.28 | 0.87±0.19 | 0.98±0.21 | 0.76±0.06 | 0.71±0.09 | 1.42±0.18 | 0.94±0.12 | 0.87±0.11 | 0.76±0.06 | 0.73±0.08 | 1.46±0.17 | 0.97±0.11 | 0.73±0.08 |
| CMAES | 0.98±0.01 | 0.98±0.01 | 1.96±0.01 | 1.31±0.01 | 1.47±0.01 | - | - | - | - | - | - | - | - | - | - |
| TTDDEA | 0.22±0.09 | -1.82±2.28 | -3.65±4.55 | -2.43±3.04 | -2.74±3.42 | 0.29±0.08 | -2.08±2.47 | -4.16±4.93 | -2.77±3.29 | -2.50±2.97 | 0.27±0.10 | -2.04±2.45 | -4.08±4.89 | -2.72±3.26 | -2.05±2.46 |
| Trimentoring | 0.92±0.03 | 0.92±0.02 | 1.81±0.04 | 1.21±0.02 | 1.37±0.02 | 0.93±0.04 | 0.91±0.02 | 1.82±0.03 | 1.21±0.02 | 1.09±0.02 | 0.92±0.04 | 0.91±0.03 | 1.83±0.05 | 1.22±0.03 | 0.92±0.03 |
| **Task** $f(x^*_{OFF})$ | Mujoco 2 $0.23_{\pm0.01}$ | | | | | | | | | | | | | | |
| ARCOO | 0.23±0.01 | 1.00±0.00 | 1.00±0.00 | 1.00±0.00 | 1.00±0.00 | 0.46±0.13 | 0.74±0.12 | 1.19±0.14 | 0.90±0.06 | 0.84±0.08 | 0.51±0.11 | 0.68±0.08 | 1.36±0.17 | 0.91±0.11 | 0.68±0.08 |
| BO | 0.55±0.20 | 0.69±0.08 | 1.22±0.08 | 0.88±0.06 | 0.97±0.06 | 0.51±0.17 | 0.62±0.04 | 1.23±0.07 | 0.82±0.05 | 0.74±0.04 | 0.43±0.13 | 0.61±0.03 | 1.22±0.07 | 0.82±0.04 | 0.61±0.03 |
| CBAS | -1.86±0.00 | -205.86±0.46 | -205.86±0.46 | -205.86±0.46 | -205.86±0.46 | -1.86±0.00 | -206.94±0.46 | -206.94±0.46 | -206.94±0.46 | -206.94±0.46 | -1.86±0.00 | -207.29±0.46 | -207.29±0.46 | -207.29±0.46 | -207.29±0.46 |
| CCDDEA | 0.17±0.20 | 0.62±0.06 | 1.23±0.11 | 0.82±0.07 | 0.93±0.08 | 0.27±0.13 | 0.57±0.07 | 1.14±0.14 | 0.76±0.09 | 0.69±0.08 | 0.27±0.13 | 0.55±0.08 | 1.11±0.17 | 0.74±0.11 | 0.56±0.08 |
| CMAES | 0.76±0.07 | 0.81±0.05 | 1.63±0.09 | 1.08±0.06 | 1.22±0.07 | - | - | - | - | - | - | - | - | - | - |
| TTDDEA | -0.22±0.12 | -5.82±12.76 | -6.70±12.51 | -6.11±12.67 | -6.26±12.63 | -0.26±0.03 | -6.21±13.39 | -7.22±13.10 | -6.54±13.29 | -6.41±13.33 | -0.23±0.07 | -6.48±13.98 | -7.54±13.67 | -6.84±13.87 | -6.49±13.98 |
| Trimentoring | 0.71±0.19 | 0.77±0.07 | 1.54±0.14 | 1.03±0.10 | 1.16±0.11 | 0.61±0.14 | 0.76±0.08 | 1.52±0.16 | 1.01±0.10 | 0.92±0.10 | 0.57±0.09 | 0.75±0.07 | 1.50±0.15 | 1.00±0.10 | 0.75±0.07 |

Table 45: Overall results for mujoco unconstrained tasks with 128 solutions and 50th percentile evaluations. In this case, 0% of the values are missing near the worst value and another 50% near the optimal value. Details are the same as Table 5.

**Mujoco 1** — $f(x^*_{OFF}) = 0.37_{\pm0.00}$

| Metric | \multicolumn{5}{c}{t = 50} | | | | | \multicolumn{5}{c}{t = 100} | | | | | \multicolumn{5}{c}{t = 150} | | | | |
|---|---|---|---|---|---|---|---|---|---|---|---|---|---|---|---|
| | FS | SI | OI | SO | SO$_\omega$ | FS | SI | OI | SO | SO$_\omega$ | FS | SI | OI | SO | SO$_\omega$ |
| ARCOO | 0.36±0.00 | 0.99±0.00 | 0.99±0.00 | 0.99±0.00 | 0.99±0.00 | 0.36±0.00 | 0.99±0.00 | 0.99±0.00 | 0.99±0.00 | 0.99±0.00 | 0.36±0.00 | 0.99±0.00 | 0.99±0.00 | 0.99±0.00 | 0.99±0.00 |
| BO | 0.37±0.04 | 0.89±0.01 | 1.00±0.00 | 0.94±0.01 | 0.96±0.01 | 0.36±0.04 | 0.89±0.01 | 1.00±0.00 | 0.94±0.01 | 0.89±0.01 | 0.37±0.03 | 0.89±0.01 | 1.00±0.00 | 0.94±0.01 | 0.89±0.01 |
| CBAS | 0.00±0.00 | -35.22±0.12 | -35.22±0.12 | -35.22±0.12 | -35.22±0.12 | 0.00±0.00 | -35.41±0.12 | -35.41±0.12 | -35.41±0.12 | -35.41±0.12 | 0.00±0.00 | -35.47±0.12 | -35.47±0.12 | -35.47±0.12 | -35.47±0.12 |
| CCDDEA | 0.75±0.09 | 0.80±0.04 | 1.52±0.07 | 1.04±0.03 | 1.17±0.03 | 0.78±0.07 | 0.86±0.03 | 1.64±0.08 | 1.13±0.02 | 1.15±0.04 | 0.78±0.07 | 0.88±0.03 | 1.68±0.10 | 1.15±0.04 | 0.88±0.03 |
| CMAES | 0.75±0.04 | 0.94±0.02 | 1.52±0.05 | 1.16±0.02 | 1.26±0.03 | - | - | - | - | - | - | - | - | - | - |
| TTDDEA | 0.22±0.09 | 0.57±0.16 | 0.61±0.21 | 0.58±0.19 | 0.59±0.20 | 0.22±0.09 | 0.52±0.23 | 0.55±0.28 | 0.53±0.25 | 0.53±0.24 | 0.22±0.09 | 0.52±0.22 | 0.55±0.27 | 0.53±0.24 | 0.52±0.22 |
| Trimentoring | 0.64±0.06 | 0.92±0.03 | 1.35±0.08 | 1.09±0.02 | 1.17±0.03 | 0.64±0.08 | 0.94±0.02 | 1.41±0.11 | 1.13±0.03 | 1.06±0.02 | 0.63±0.09 | 0.95±0.03 | 1.42±0.12 | 1.14±0.04 | 0.95±0.03 |

**Mujoco 2** — $f(x^*_{OFF}) = -0.10_{\pm0.00}$

| Metric | FS | SI | OI | SO | SO$_\omega$ | FS | SI | OI | SO | SO$_\omega$ | FS | SI | OI | SO | SO$_\omega$ |
|---|---|---|---|---|---|---|---|---|---|---|---|---|---|---|---|
| ARCOO | -0.10±0.00 | 0.99±0.00 | 0.99±0.00 | 0.99±0.00 | 0.99±0.00 | -0.10±0.00 | 0.99±0.00 | 0.99±0.00 | 0.99±0.00 | 0.99±0.00 | -0.10±0.00 | 0.99±0.00 | 0.99±0.00 | 0.99±0.00 | 0.99±0.00 |
| BO | -0.12±0.06 | 0.91±0.03 | 1.00±0.01 | 0.95±0.02 | 0.97±0.01 | -0.11±0.04 | 0.89±0.04 | 1.00±0.01 | 0.94±0.02 | 0.93±0.03 | -0.10±0.06 | 0.89±0.04 | 1.00±0.01 | 0.94±0.02 | 0.89±0.04 |
| CBAS | -1.86±0.00 | -173.04±0.27 | -173.04±0.27 | -173.04±0.27 | -173.04±0.27 | -1.86±0.00 | -173.95±0.27 | -173.95±0.27 | -173.95±0.27 | -173.95±0.27 | -1.86±0.00 | -174.25±0.27 | -174.25±0.27 | -174.25±0.27 | -174.25±0.27 |
| CCDDEA | 0.06±0.29 | 0.78±0.07 | 1.39±0.11 | 1.00±0.07 | 1.10±0.08 | 0.25±0.11 | 0.80±0.06 | 1.44±0.11 | 1.02±0.06 | 0.94±0.06 | 0.25±0.11 | 0.82±0.06 | 1.48±0.13 | 1.05±0.07 | 0.82±0.06 |
| CMAES | 0.18±0.03 | 0.97±0.01 | 1.29±0.02 | 1.11±0.01 | 1.16±0.01 | - | - | - | - | - | - | - | - | - | - |
| TTDDEA | -0.44±0.73 | 0.71±0.28 | 0.89±0.36 | 0.79±0.31 | 0.82±0.33 | -0.45±0.74 | 0.68±0.28 | 0.86±0.36 | 0.76±0.31 | 0.73±0.30 | -0.46±0.73 | 0.67±0.28 | 0.85±0.37 | 0.75±0.32 | 0.67±0.28 |
| Trimentoring | 0.23±0.05 | 0.91±0.03 | 1.30±0.08 | 1.07±0.01 | 1.14±0.03 | 0.22±0.05 | 0.95±0.02 | 1.37±0.11 | 1.12±0.03 | 1.06±0.01 | 0.23±0.05 | 0.96±0.01 | 1.39±0.12 | 1.14±0.03 | 0.96±0.01 |

Table 46: Overall results for mujoco unconstrained tasks with 128 solutions and 50th percentile evaluations. In this case, 0% of the values are missing near the worst value and another 40% near the optimal value. Details are the same as Table 5.

**Mujoco 1** — $f(x^*_{OFF}) = 0.40_{\pm0.00}$

| Metric | FS | SI | OI | SO | SO$_\omega$ | FS | SI | OI | SO | SO$_\omega$ | FS | SI | OI | SO | SO$_\omega$ |
|---|---|---|---|---|---|---|---|---|---|---|---|---|---|---|---|
| ARCOO | 0.40±0.00 | 0.99±0.01 | 0.99±0.01 | 0.99±0.01 | 0.99±0.01 | 0.40±0.00 | 0.99±0.01 | 0.99±0.01 | 0.99±0.01 | 0.99±0.01 | 0.40±0.00 | 0.99±0.01 | 0.99±0.01 | 0.99±0.01 | 0.99±0.01 |
| BO | 0.37±0.03 | 0.88±0.01 | 0.94±0.01 | 0.91±0.01 | 0.92±0.01 | 0.38±0.03 | 0.86±0.01 | 0.94±0.01 | 0.90±0.01 | 0.89±0.01 | 0.37±0.03 | 0.86±0.01 | 0.94±0.00 | 0.90±0.01 | 0.86±0.01 |
| CBAS | 0.00±0.00 | -38.87±0.22 | -38.87±0.22 | -38.87±0.22 | -38.87±0.22 | 0.00±0.00 | -39.08±0.22 | -39.08±0.22 | -39.08±0.22 | -39.08±0.22 | 0.00±0.00 | -39.15±0.22 | -39.15±0.22 | -39.15±0.22 | -39.15±0.22 |
| CCDDEA | 0.73±0.16 | 0.77±0.08 | 1.49±0.15 | 1.02±0.10 | 1.14±0.11 | 0.77±0.15 | 0.83±0.09 | 1.61±0.15 | 1.10±0.10 | 1.00±0.10 | 0.77±0.15 | 0.85±0.10 | 1.65±0.18 | 1.12±0.12 | 0.86±0.10 |
| CMAES | 0.79±0.03 | 0.93±0.01 | 1.55±0.04 | 1.16±0.01 | 1.27±0.01 | - | - | - | - | - | - | - | - | - | - |
| TTDDEA | 0.22±0.10 | 0.52±0.15 | 0.55±0.18 | 0.53±0.17 | 0.54±0.17 | 0.21±0.10 | 0.47±0.19 | 0.50±0.23 | 0.49±0.21 | 0.48±0.20 | 0.21±0.10 | 0.45±0.21 | 0.48±0.24 | 0.46±0.22 | 0.45±0.21 |
| Trimentoring | 0.61±0.12 | 0.86±0.08 | 1.19±0.23 | 1.00±0.13 | 1.05±0.16 | 0.63±0.12 | 0.88±0.09 | 1.27±0.26 | 1.03±0.15 | 0.98±0.13 | 0.62±0.13 | 0.90±0.10 | 1.30±0.27 | 1.06±0.16 | 0.90±0.10 |

**Mujoco 2** — $f(x^*_{OFF}) = -0.04_{\pm0.00}$

| Metric | FS | SI | OI | SO | SO$_\omega$ | FS | SI | OI | SO | SO$_\omega$ | FS | SI | OI | SO | SO$_\omega$ |
|---|---|---|---|---|---|---|---|---|---|---|---|---|---|---|---|
| ARCOO | -0.04±0.00 | 0.99±0.00 | 0.99±0.00 | 0.99±0.00 | 0.99±0.00 | -0.04±0.00 | 0.99±0.01 | 0.99±0.00 | 0.99±0.01 | 0.99±0.01 | -0.04±0.01 | 0.99±0.01 | 0.99±0.01 | 0.99±0.01 | 0.99±0.01 |
| BO | -0.08±0.03 | 0.91±0.04 | 0.95±0.02 | 0.93±0.03 | 0.93±0.02 | -0.10±0.04 | 0.89±0.04 | 0.95±0.02 | 0.92±0.03 | 0.91±0.03 | -0.11±0.06 | 0.89±0.03 | 0.95±0.01 | 0.91±0.02 | 0.89±0.03 |
| CBAS | -1.86±0.00 | -179.29±0.34 | -179.29±0.34 | -179.29±0.34 | -179.29±0.34 | -1.86±0.00 | -180.23±0.34 | -180.23±0.34 | -180.23±0.34 | -180.23±0.34 | -1.86±0.00 | -180.54±0.34 | -180.54±0.34 | -180.54±0.34 | -180.54±0.34 |
| CCDDEA | 0.10±0.13 | 0.78±0.04 | 1.42±0.10 | 1.00±0.05 | 1.11±0.06 | 0.32±0.12 | 0.79±0.04 | 1.47±0.11 | 1.03±0.06 | 0.94±0.05 | 0.32±0.12 | 0.81±0.05 | 1.52±0.16 | 1.06±0.08 | 0.82±0.05 |
| CMAES | 0.18±0.02 | 0.97±0.00 | 1.21±0.03 | 1.08±0.01 | 1.12±0.02 | - | - | - | - | - | - | - | - | - | - |
| TTDDEA | -0.83±0.84 | 0.58±0.33 | 0.64±0.35 | 0.61±0.34 | 0.62±0.34 | -0.41±0.62 | 0.53±0.37 | 0.58±0.42 | 0.55±0.39 | 0.54±0.39 | -0.54±0.60 | 0.57±0.30 | 0.65±0.33 | 0.61±0.31 | 0.57±0.30 |
| Trimentoring | 0.15±0.12 | 0.87±0.07 | 1.18±0.21 | 0.99±0.11 | 1.04±0.14 | 0.15±0.12 | 0.89±0.07 | 1.22±0.24 | 1.02±0.13 | 0.97±0.10 | 0.14±0.12 | 0.90±0.08 | 1.23±0.26 | 1.03±0.14 | 0.90±0.08 |

Table 47: Overall results for mujoco unconstrained tasks with 128 solutions and 50th percentile evaluations. In this case, 0% of the values are missing near the worst value and another 30% near the optimal value. Details are the same as Table 5.

**Mujoco 1** — $f(x^*_{OFF}) = 0.45_{\pm0.00}$

| Metric | FS | SI | OI | SO | SO$_\omega$ | FS | SI | OI | SO | SO$_\omega$ | FS | SI | OI | SO | SO$_\omega$ |
|---|---|---|---|---|---|---|---|---|---|---|---|---|---|---|---|
| ARCOO | 0.44±0.00 | 0.99±0.01 | 0.99±0.01 | 0.99±0.01 | 0.99±0.01 | 0.44±0.00 | 0.99±0.01 | 0.99±0.01 | 0.99±0.01 | 0.99±0.01 | 0.44±0.00 | 0.99±0.01 | 0.99±0.01 | 0.99±0.01 | 0.99±0.01 |
| BO | 0.37±0.03 | 0.82±0.09 | 0.86±0.01 | 0.83±0.05 | 0.84±0.03 | 0.36±0.03 | 0.81±0.09 | 0.85±0.01 | 0.83±0.05 | 0.83±0.05 | 0.36±0.02 | 0.81±0.09 | 0.86±0.01 | 0.83±0.05 | 0.81±0.09 |
| CBAS | 0.00±0.00 | -43.12±0.18 | -43.12±0.18 | -43.12±0.18 | -43.12±0.18 | 0.00±0.00 | -43.35±0.18 | -43.35±0.18 | -43.35±0.18 | -43.35±0.18 | 0.00±0.00 | -43.42±0.18 | -43.42±0.18 | -43.42±0.18 | -43.42±0.18 |
| CCDDEA | 0.73±0.07 | 0.72±0.08 | 1.37±0.17 | 0.94±0.10 | 1.05±0.12 | 0.73±0.08 | 0.79±0.07 | 1.51±0.14 | 1.03±0.08 | 0.94±0.08 | 0.73±0.08 | 0.81±0.08 | 1.55±0.15 | 1.06±0.09 | 0.81±0.08 |
| CMAES | 0.81±0.04 | 0.92±0.01 | 1.54±0.05 | 1.15±0.01 | 1.26±0.02 | - | - | - | - | - | - | - | - | - | - |
| TTDDEA | 0.20±0.20 | 0.10±0.20 | 0.10±0.20 | 0.10±0.20 | 0.10±0.20 | 0.23±0.07 | 0.13±0.26 | 0.13±0.26 | 0.13±0.26 | 0.13±0.26 | 0.23±0.07 | 0.15±0.28 | 0.15±0.28 | 0.15±0.28 | 0.15±0.28 |
| Trimentoring | 0.67±0.09 | 0.91±0.02 | 1.33±0.10 | 1.08±0.03 | 1.16±0.05 | 0.70±0.06 | 0.93±0.03 | 1.40±0.12 | 1.12±0.05 | 1.05±0.04 | 0.70±0.06 | 0.94±0.03 | 1.43±0.13 | 1.13±0.05 | 0.94±0.03 |

**Mujoco 2** — $f(x^*_{OFF}) = 0.03_{\pm0.01}$

| Metric | FS | SI | OI | SO | SO$_\omega$ | FS | SI | OI | SO | SO$_\omega$ | FS | SI | OI | SO | SO$_\omega$ |
|---|---|---|---|---|---|---|---|---|---|---|---|---|---|---|---|
| ARCOO | 0.02±0.00 | 0.99±0.01 | 0.99±0.01 | 0.99±0.01 | 0.99±0.01 | 0.02±0.00 | 0.99±0.01 | 0.99±0.01 | 0.99±0.01 | 0.99±0.01 | 0.02±0.00 | 0.99±0.01 | 0.99±0.01 | 0.99±0.01 | 0.99±0.01 |
| BO | -0.07±0.06 | 0.89±0.05 | 0.91±0.03 | 0.90±0.04 | 0.90±0.03 | -0.13±0.06 | 0.86±0.08 | 0.91±0.02 | 0.88±0.05 | 0.88±0.06 | -0.10±0.06 | 0.86±0.08 | 0.91±0.03 | 0.88±0.05 | 0.86±0.08 |
| CBAS | -1.86±0.00 | -185.68±0.45 | -185.68±0.45 | -185.68±0.45 | -185.68±0.45 | -1.86±0.00 | -186.65±0.46 | -186.65±0.46 | -186.65±0.46 | -186.65±0.46 | -1.86±0.00 | -186.97±0.46 | -186.97±0.46 | -186.97±0.46 | -186.97±0.46 |
| CCDDEA | 0.12±0.29 | 0.76±0.05 | 1.43±0.08 | 0.99±0.04 | 1.10±0.04 | 0.34±0.14 | 0.76±0.04 | 1.45±0.09 | 0.99±0.06 | 0.90±0.06 | 0.34±0.14 | 0.76±0.08 | 1.46±0.13 | 1.00±0.09 | 0.77±0.08 |
| CMAES | 0.20±0.03 | 0.97±0.01 | 1.16±0.04 | 1.05±0.01 | 1.09±0.02 | - | - | - | - | - | - | - | - | - | - |
| TTDDEA | -1.86±0.00 | 0.09±0.28 | 0.09±0.28 | 0.09±0.28 | 0.09±0.28 | -1.55±0.57 | 0.03±0.29 | 0.03±0.29 | 0.03±0.29 | 0.03±0.29 | -1.65±0.54 | 0.05±0.30 | 0.05±0.30 | 0.05±0.30 | 0.05±0.30 |
| Trimentoring | 0.22±0.05 | 0.90±0.03 | 1.17±0.06 | 1.02±0.03 | 1.06±0.04 | 0.22±0.04 | 0.94±0.02 | 1.22±0.07 | 1.06±0.03 | 1.02±0.03 | 0.22±0.05 | 0.95±0.03 | 1.24±0.08 | 1.07±0.04 | 0.95±0.03 |

Table 48: Overall results for mujoco unconstrained tasks with 128 solutions and 50th percentile evaluations. In this case, 0% of the values are missing near the worst value and another 20% near the optimal value. Details are the same as Table 5.

**Mujoco 1** — $f(x^*_{OFF}) = 0.50_{\pm0.00}$

| Metric | FS | SI | OI | SO | SO$_\omega$ | FS | SI | OI | SO | SO$_\omega$ | FS | SI | OI | SO | SO$_\omega$ |
|---|---|---|---|---|---|---|---|---|---|---|---|---|---|---|---|
| ARCOO | 0.50±0.00 | 0.98±0.00 | 0.98±0.00 | 0.98±0.00 | 0.98±0.00 | 0.50±0.00 | 0.98±0.00 | 0.98±0.00 | 0.98±0.00 | 0.98±0.00 | 0.49±0.00 | 0.98±0.01 | 0.98±0.01 | 0.98±0.01 | 0.98±0.01 |
| BO | 0.38±0.04 | 0.68±0.09 | 0.73±0.01 | 0.70±0.05 | 0.71±0.03 | 0.37±0.03 | 0.68±0.09 | 0.73±0.01 | 0.70±0.05 | 0.70±0.05 | 0.35±0.03 | 0.68±0.09 | 0.73±0.01 | 0.70±0.05 | 0.68±0.09 |
| CBAS | 0.00±0.00 | -48.64±0.34 | -48.64±0.34 | -48.64±0.34 | -48.64±0.34 | 0.00±0.00 | -48.89±0.34 | -48.89±0.34 | -48.89±0.34 | -48.89±0.34 | 0.00±0.00 | -48.98±0.34 | -48.98±0.34 | -48.98±0.34 | -48.98±0.34 |
| CCDDEA | 0.75±0.11 | 0.74±0.05 | 1.40±0.14 | 0.96±0.07 | 1.08±0.08 | 0.75±0.11 | 0.78±0.06 | 1.48±0.16 | 1.01±0.08 | 0.92±0.07 | 0.75±0.11 | 0.79±0.07 | 1.50±0.18 | 1.03±0.10 | 0.79±0.07 |
| CMAES | 0.81±0.04 | 0.91±0.02 | 1.49±0.04 | 1.13±0.02 | 1.23±0.02 | - | - | - | - | - | - | - | - | - | - |
| TTDDEA | 0.27±0.11 | -0.22±0.64 | -0.22±0.64 | -0.22±0.64 | -0.22±0.64 | 0.27±0.11 | -0.28±0.68 | -0.28±0.68 | -0.28±0.68 | -0.28±0.68 | 0.27±0.11 | -0.30±0.71 | -0.30±0.71 | -0.30±0.71 | -0.30±0.71 |
| Trimentoring | 0.70±0.04 | 0.90±0.02 | 1.30±0.07 | 1.06±0.03 | 1.13±0.04 | 0.70±0.04 | 0.94±0.02 | 1.36±0.08 | 1.11±0.03 | 1.05±0.02 | 0.70±0.05 | 0.95±0.02 | 1.38±0.08 | 1.13±0.03 | 0.95±0.02 |

**Mujoco 2** — $f(x^*_{OFF}) = 0.11_{\pm0.00}$

| Metric | FS | SI | OI | SO | SO$_\omega$ | FS | SI | OI | SO | SO$_\omega$ | FS | SI | OI | SO | SO$_\omega$ |
|---|---|---|---|---|---|---|---|---|---|---|---|---|---|---|---|
| ARCOO | 0.11±0.00 | 0.99±0.01 | 0.99±0.01 | 0.99±0.01 | 0.99±0.01 | 0.11±0.00 | 0.99±0.01 | 0.99±0.01 | 0.99±0.01 | 0.99±0.01 | 0.11±0.00 | 0.99±0.01 | 0.99±0.01 | 0.99±0.01 | 0.99±0.01 |
| BO | -0.10±0.04 | 0.80±0.04 | 0.80±0.04 | 0.80±0.04 | 0.80±0.04 | -0.11±0.07 | 0.79±0.04 | 0.79±0.04 | 0.79±0.04 | 0.79±0.04 | -0.09±0.03 | 0.79±0.04 | 0.79±0.04 | 0.79±0.04 | 0.79±0.04 |
| CBAS | -1.86±0.00 | -194.21±0.35 | -194.21±0.35 | -194.21±0.35 | -194.21±0.35 | -1.86±0.00 | -195.23±0.35 | -195.23±0.35 | -195.23±0.35 | -195.23±0.35 | -1.86±0.00 | -195.56±0.35 | -195.56±0.35 | -195.56±0.35 | -195.56±0.35 |
| CCDDEA | 0.12±0.17 | 0.72±0.03 | 1.32±0.08 | 0.93±0.04 | 1.04±0.05 | 0.38±0.09 | 0.72±0.06 | 1.39±0.07 | 0.94±0.06 | 0.85±0.06 | 0.38±0.09 | 0.74±0.08 | 1.44±0.10 | 0.98±0.09 | 0.74±0.08 |
| CMAES | 0.23±0.04 | 0.96±0.01 | 1.14±0.04 | 1.04±0.02 | 1.07±0.02 | - | - | - | - | - | - | - | - | - | - |
| TTDDEA | -1.86±0.00 | -0.98±0.58 | -0.98±0.58 | -0.98±0.58 | -0.98±0.58 | -1.86±0.00 | -1.07±0.60 | -1.07±0.60 | -1.07±0.60 | -1.07±0.60 | -1.86±0.00 | -1.10±0.60 | -1.10±0.60 | -1.10±0.60 | -1.10±0.60 |
| Trimentoring | 0.21±0.03 | 0.93±0.03 | 1.08±0.04 | 1.00±0.02 | 1.03±0.02 | 0.21±0.03 | 0.94±0.01 | 1.12±0.06 | 1.03±0.02 | 1.00±0.02 | 0.21±0.03 | 0.95±0.02 | 1.14±0.06 | 1.03±0.03 | 0.95±0.02 |

Table 49: Overall results for mujoco unconstrained tasks with 128 solutions and 50th percentile evaluations. In this case, 0% of the values are missing near the worst value and another 10% near the optimal value. Details are the same as Table 5.

| Steps | | t = 50 | | | | | | t = 100 | | | | | | t = 150 | | | |
|---|---|---|---|---|---|---|---|---|---|---|---|---|---|---|---|---|---|
| Task $f(x^*_{OFF})$ | | | | | | | | Mujoco 1 $0.59_{\pm 0.01}$ | | | | | | | | | |
| Metric | FS | SI | OI | SO | SO$_\omega$ | FS | SI | OI | SO | SO$_\omega$ | FS | SI | OI | SO | SO$_\omega$ | | |
| ARCOO | $0.58_{\pm 0.01}$ | $0.96_{\pm 0.01}$ | $0.96_{\pm 0.01}$ | $0.96_{\pm 0.01}$ | $0.96_{\pm 0.01}$ | $0.58_{\pm 0.01}$ | $0.96_{\pm 0.01}$ | $0.96_{\pm 0.01}$ | $0.96_{\pm 0.01}$ | $0.96_{\pm 0.01}$ | $0.58_{\pm 0.01}$ | $0.96_{\pm 0.01}$ | $0.96_{\pm 0.01}$ | $0.96_{\pm 0.01}$ | $0.96_{\pm 0.01}$ | | |
| BO | $0.38_{\pm 0.03}$ | $0.40_{\pm 0.05}$ | $0.45_{\pm 0.03}$ | $0.42_{\pm 0.03}$ | $0.43_{\pm 0.02}$ | $0.37_{\pm 0.03}$ | $0.39_{\pm 0.05}$ | $0.44_{\pm 0.03}$ | $0.41_{\pm 0.03}$ | $0.40_{\pm 0.04}$ | $0.38_{\pm 0.02}$ | $0.38_{\pm 0.05}$ | $0.43_{\pm 0.02}$ | $0.40_{\pm 0.03}$ | $0.38_{\pm 0.05}$ | | |
| CBAS | $0.00_{\pm 0.00}$ | $-57.79_{\pm 0.58}$ | $-57.79_{\pm 0.58}$ | $-57.79_{\pm 0.58}$ | $-57.79_{\pm 0.58}$ | $0.00_{\pm 0.00}$ | $-58.10_{\pm 0.58}$ | $-58.10_{\pm 0.58}$ | $-58.10_{\pm 0.58}$ | $-58.10_{\pm 0.58}$ | $0.00_{\pm 0.00}$ | $-58.20_{\pm 0.59}$ | $-58.20_{\pm 0.59}$ | $-58.20_{\pm 0.59}$ | $-58.20_{\pm 0.59}$ | | |
| CCDDEA | $0.71_{\pm 0.07}$ | $0.68_{\pm 0.10}$ | $1.25_{\pm 0.21}$ | $0.88_{\pm 0.13}$ | $0.97_{\pm 0.15}$ | $0.70_{\pm 0.06}$ | $0.70_{\pm 0.08}$ | $1.29_{\pm 0.09}$ | $0.91_{\pm 0.08}$ | $0.83_{\pm 0.08}$ | $0.70_{\pm 0.06}$ | $0.71_{\pm 0.10}$ | $1.31_{\pm 0.10}$ | $0.92_{\pm 0.10}$ | $0.71_{\pm 0.10}$ | | |
| CMAES | $0.82_{\pm 0.04}$ | $0.90_{\pm 0.01}$ | $1.45_{\pm 0.06}$ | $1.11_{\pm 0.02}$ | $1.21_{\pm 0.03}$ | - | - | - | - | - | - | - | - | - | - | | |
| TTDDEA | $0.34_{\pm 0.12}$ | $-4.71_{\pm 4.11}$ | $-4.71_{\pm 4.11}$ | $-4.71_{\pm 4.11}$ | $-4.71_{\pm 4.11}$ | $0.34_{\pm 0.12}$ | $-4.82_{\pm 4.52}$ | $-4.82_{\pm 4.52}$ | $-4.82_{\pm 4.52}$ | $-4.82_{\pm 4.52}$ | $0.34_{\pm 0.12}$ | $-4.81_{\pm 4.69}$ | $-4.81_{\pm 4.69}$ | $-4.81_{\pm 4.69}$ | $-4.81_{\pm 4.69}$ | | |
| Trimentoring | $0.72_{\pm 0.07}$ | $0.90_{\pm 0.04}$ | $1.22_{\pm 0.14}$ | $1.03_{\pm 0.05}$ | $1.09_{\pm 0.07}$ | $0.72_{\pm 0.08}$ | $0.93_{\pm 0.03}$ | $1.28_{\pm 0.17}$ | $1.07_{\pm 0.06}$ | $1.02_{\pm 0.04}$ | $0.72_{\pm 0.06}$ | $0.94_{\pm 0.02}$ | $1.29_{\pm 0.18}$ | $1.08_{\pm 0.07}$ | $0.94_{\pm 0.02}$ | | |
| Task $f(x^*_{OFF})$ | | | | | | | | Mujoco 2 $0.23_{\pm 0.01}$ | | | | | | | | | |
| ARCOO | $0.22_{\pm 0.01}$ | $0.97_{\pm 0.02}$ | $0.97_{\pm 0.02}$ | $0.97_{\pm 0.02}$ | $0.97_{\pm 0.02}$ | $0.22_{\pm 0.01}$ | $0.97_{\pm 0.02}$ | $0.97_{\pm 0.02}$ | $0.97_{\pm 0.02}$ | $0.97_{\pm 0.02}$ | $0.22_{\pm 0.00}$ | $0.97_{\pm 0.02}$ | $0.97_{\pm 0.02}$ | $0.97_{\pm 0.02}$ | $0.97_{\pm 0.02}$ | | |
| BO | $-0.11_{\pm 0.02}$ | $0.68_{\pm 0.07}$ | $0.68_{\pm 0.07}$ | $0.68_{\pm 0.07}$ | $0.68_{\pm 0.07}$ | $-0.08_{\pm 0.02}$ | $0.67_{\pm 0.07}$ | $0.67_{\pm 0.07}$ | $0.67_{\pm 0.07}$ | $0.67_{\pm 0.07}$ | $-0.10_{\pm 0.04}$ | $0.67_{\pm 0.08}$ | $0.67_{\pm 0.08}$ | $0.67_{\pm 0.08}$ | $0.67_{\pm 0.08}$ | | |
| CBAS | $-1.86_{\pm 0.00}$ | $-205.86_{\pm 0.46}$ | $-205.86_{\pm 0.46}$ | $-205.86_{\pm 0.46}$ | $-205.86_{\pm 0.46}$ | $-1.86_{\pm 0.00}$ | $-206.94_{\pm 0.46}$ | $-206.94_{\pm 0.46}$ | $-206.94_{\pm 0.46}$ | $-206.94_{\pm 0.46}$ | $-1.86_{\pm 0.00}$ | $-207.29_{\pm 0.46}$ | $-207.29_{\pm 0.46}$ | $-207.29_{\pm 0.46}$ | $-207.29_{\pm 0.46}$ | | |
| CCDDEA | $0.00_{\pm 0.23}$ | $0.66_{\pm 0.07}$ | $1.21_{\pm 0.17}$ | $0.85_{\pm 0.10}$ | $0.94_{\pm 0.12}$ | $0.38_{\pm 0.11}$ | $0.68_{\pm 0.07}$ | $1.25_{\pm 0.11}$ | $0.88_{\pm 0.09}$ | $0.80_{\pm 0.08}$ | $0.38_{\pm 0.11}$ | $0.70_{\pm 0.10}$ | $1.29_{\pm 0.15}$ | $0.91_{\pm 0.12}$ | $0.70_{\pm 0.10}$ | | |
| CMAES | $0.25_{\pm 0.03}$ | $0.96_{\pm 0.02}$ | $1.01_{\pm 0.03}$ | $0.98_{\pm 0.02}$ | $0.99_{\pm 0.02}$ | - | - | - | - | - | - | - | - | - | - | | |
| TTDDEA | $-0.93_{\pm 0.93}$ | $-1.28_{\pm 1.85}$ | $-1.28_{\pm 1.85}$ | $-1.28_{\pm 1.85}$ | $-1.28_{\pm 1.85}$ | $-0.95_{\pm 0.91}$ | $-1.33_{\pm 2.10}$ | $-1.33_{\pm 2.10}$ | $-1.33_{\pm 2.10}$ | $-1.33_{\pm 2.10}$ | $-0.95_{\pm 0.91}$ | $-1.35_{\pm 2.18}$ | $-1.35_{\pm 2.18}$ | $-1.35_{\pm 2.18}$ | $-1.35_{\pm 2.18}$ | | |
| Trimentoring | $0.24_{\pm 0.02}$ | $0.93_{\pm 0.02}$ | $0.96_{\pm 0.03}$ | $0.95_{\pm 0.02}$ | $0.95_{\pm 0.02}$ | $0.24_{\pm 0.03}$ | $0.94_{\pm 0.02}$ | $0.99_{\pm 0.02}$ | $0.96_{\pm 0.02}$ | $0.95_{\pm 0.02}$ | $0.24_{\pm 0.04}$ | $0.94_{\pm 0.02}$ | $0.99_{\pm 0.03}$ | $0.97_{\pm 0.02}$ | $0.94_{\pm 0.02}$ | | |

Table 50: Overall results for mujoco unconstrained tasks with 128 solutions and 50th percentile evaluations. In this case, 10% of the values are missing near the worst value and another 40% near the optimal value. Details are the same as Table 5.

| Steps | | t = 50 | | | | | | t = 100 | | | | | | t = 150 | | |
|---|---|---|---|---|---|---|---|---|---|---|---|---|---|---|---|---|
| Task $f(x^*_{OFF})$ | | | | | | | | Mujoco 1 $0.40_{\pm 0.00}$ | | | | | | | | |
| Metric | FS | SI | OI | SO | SO$_\omega$ | FS | SI | OI | SO | SO$_\omega$ | FS | SI | OI | SO | SO$_\omega$ |
| ARCOO | $0.40_{\pm 0.00}$ | $0.99_{\pm 0.00}$ | $0.99_{\pm 0.00}$ | $0.99_{\pm 0.00}$ | $0.99_{\pm 0.00}$ | $0.40_{\pm 0.00}$ | $0.99_{\pm 0.00}$ | $0.99_{\pm 0.00}$ | $0.99_{\pm 0.00}$ | $0.99_{\pm 0.00}$ | $0.40_{\pm 0.00}$ | $0.99_{\pm 0.00}$ | $0.99_{\pm 0.00}$ | $0.99_{\pm 0.00}$ | $0.99_{\pm 0.00}$ |
| BO | $0.38_{\pm 0.03}$ | $0.89_{\pm 0.01}$ | $0.94_{\pm 0.01}$ | $0.91_{\pm 0.01}$ | $0.92_{\pm 0.01}$ | $0.37_{\pm 0.03}$ | $0.88_{\pm 0.01}$ | $0.94_{\pm 0.01}$ | $0.91_{\pm 0.01}$ | $0.90_{\pm 0.01}$ | $0.36_{\pm 0.03}$ | $0.87_{\pm 0.02}$ | $0.94_{\pm 0.00}$ | $0.90_{\pm 0.01}$ | $0.87_{\pm 0.02}$ |
| CBAS | $0.00_{\pm 0.00}$ | $-38.87_{\pm 0.22}$ | $-38.87_{\pm 0.22}$ | $-38.87_{\pm 0.22}$ | $-38.87_{\pm 0.22}$ | $0.00_{\pm 0.00}$ | $-39.08_{\pm 0.22}$ | $-39.08_{\pm 0.22}$ | $-39.08_{\pm 0.22}$ | $-39.08_{\pm 0.22}$ | $0.00_{\pm 0.00}$ | $-39.15_{\pm 0.22}$ | $-39.15_{\pm 0.22}$ | $-39.15_{\pm 0.22}$ | $-39.15_{\pm 0.22}$ |
| CCDDEA | $0.71_{\pm 0.20}$ | $0.74_{\pm 0.04}$ | $1.42_{\pm 0.12}$ | $0.97_{\pm 0.06}$ | $1.09_{\pm 0.07}$ | $0.74_{\pm 0.10}$ | $0.80_{\pm 0.04}$ | $1.53_{\pm 0.13}$ | $1.05_{\pm 0.06}$ | $0.95_{\pm 0.05}$ | $0.74_{\pm 0.10}$ | $0.82_{\pm 0.05}$ | $1.57_{\pm 0.15}$ | $1.07_{\pm 0.07}$ | $0.82_{\pm 0.05}$ |
| CMAES | $0.74_{\pm 0.07}$ | $0.93_{\pm 0.02}$ | $1.49_{\pm 0.09}$ | $1.15_{\pm 0.03}$ | $1.24_{\pm 0.04}$ | - | - | - | - | - | - | - | - | - | - |
| TTDDEA | $0.21_{\pm 0.10}$ | $0.50_{\pm 0.33}$ | $0.50_{\pm 0.33}$ | $0.50_{\pm 0.33}$ | $0.50_{\pm 0.33}$ | $0.21_{\pm 0.10}$ | $0.48_{\pm 0.38}$ | $0.48_{\pm 0.38}$ | $0.48_{\pm 0.38}$ | $0.48_{\pm 0.38}$ | $0.21_{\pm 0.10}$ | $0.47_{\pm 0.39}$ | $0.47_{\pm 0.39}$ | $0.47_{\pm 0.39}$ | $0.47_{\pm 0.39}$ |
| Trimentoring | $0.62_{\pm 0.07}$ | $0.91_{\pm 0.03}$ | $1.29_{\pm 0.08}$ | $1.06_{\pm 0.04}$ | $1.13_{\pm 0.05}$ | $0.63_{\pm 0.07}$ | $0.92_{\pm 0.03}$ | $1.34_{\pm 0.09}$ | $1.09_{\pm 0.04}$ | $1.03_{\pm 0.04}$ | $0.62_{\pm 0.07}$ | $0.93_{\pm 0.03}$ | $1.36_{\pm 0.10}$ | $1.10_{\pm 0.04}$ | $0.93_{\pm 0.03}$ |
| Task $f(x^*_{OFF})$ | | | | | | | | Mujoco 2 $-0.04_{\pm 0.00}$ | | | | | | | | |
| ARCOO | $-0.04_{\pm 0.00}$ | $0.99_{\pm 0.00}$ | $0.99_{\pm 0.00}$ | $0.99_{\pm 0.00}$ | $0.99_{\pm 0.00}$ | $-0.04_{\pm 0.00}$ | $0.99_{\pm 0.00}$ | $0.99_{\pm 0.00}$ | $0.99_{\pm 0.00}$ | $0.99_{\pm 0.00}$ | $-0.04_{\pm 0.01}$ | $0.99_{\pm 0.00}$ | $0.99_{\pm 0.00}$ | $0.99_{\pm 0.00}$ | $0.99_{\pm 0.00}$ |
| BO | $-0.06_{\pm 0.06}$ | $0.88_{\pm 0.03}$ | $0.94_{\pm 0.01}$ | $0.91_{\pm 0.02}$ | $0.92_{\pm 0.02}$ | $-0.09_{\pm 0.03}$ | $0.86_{\pm 0.04}$ | $0.94_{\pm 0.01}$ | $0.90_{\pm 0.03}$ | $0.89_{\pm 0.03}$ | $-0.07_{\pm 0.05}$ | $0.86_{\pm 0.04}$ | $0.94_{\pm 0.01}$ | $0.90_{\pm 0.03}$ | $0.86_{\pm 0.04}$ |
| CBAS | $-1.86_{\pm 0.00}$ | $-179.29_{\pm 0.34}$ | $-179.29_{\pm 0.34}$ | $-179.29_{\pm 0.34}$ | $-179.29_{\pm 0.34}$ | $-1.86_{\pm 0.00}$ | $-180.23_{\pm 0.34}$ | $-180.23_{\pm 0.34}$ | $-180.23_{\pm 0.34}$ | $-180.23_{\pm 0.34}$ | $-1.86_{\pm 0.00}$ | $-180.54_{\pm 0.34}$ | $-180.54_{\pm 0.34}$ | $-180.54_{\pm 0.34}$ | $-180.54_{\pm 0.34}$ |
| CCDDEA | $0.26_{\pm 0.17}$ | $0.80_{\pm 0.05}$ | $1.52_{\pm 0.10}$ | $1.04_{\pm 0.06}$ | $1.17_{\pm 0.07}$ | $0.26_{\pm 0.13}$ | $0.78_{\pm 0.06}$ | $1.49_{\pm 0.12}$ | $1.02_{\pm 0.08}$ | $0.93_{\pm 0.07}$ | $0.26_{\pm 0.13}$ | $0.77_{\pm 0.08}$ | $1.48_{\pm 0.15}$ | $1.01_{\pm 0.10}$ | $0.77_{\pm 0.08}$ |
| CMAES | $0.19_{\pm 0.03}$ | $0.97_{\pm 0.01}$ | $1.22_{\pm 0.03}$ | $1.08_{\pm 0.01}$ | $1.12_{\pm 0.02}$ | - | - | - | - | - | - | - | - | - | - |
| TTDDEA | $-0.44_{\pm 0.37}$ | $0.38_{\pm 0.22}$ | $0.44_{\pm 0.29}$ | $0.40_{\pm 0.24}$ | $0.41_{\pm 0.25}$ | $-0.38_{\pm 0.36}$ | $0.38_{\pm 0.26}$ | $0.46_{\pm 0.39}$ | $0.41_{\pm 0.30}$ | $0.40_{\pm 0.28}$ | $-0.42_{\pm 0.35}$ | $0.40_{\pm 0.27}$ | $0.48_{\pm 0.42}$ | $0.43_{\pm 0.32}$ | $0.40_{\pm 0.27}$ |
| Trimentoring | $0.19_{\pm 0.05}$ | $0.91_{\pm 0.05}$ | $1.18_{\pm 0.03}$ | $1.02_{\pm 0.03}$ | $1.07_{\pm 0.03}$ | $0.20_{\pm 0.06}$ | $0.93_{\pm 0.04}$ | $1.24_{\pm 0.06}$ | $1.06_{\pm 0.02}$ | $1.02_{\pm 0.02}$ | $0.20_{\pm 0.06}$ | $0.95_{\pm 0.02}$ | $1.26_{\pm 0.07}$ | $1.08_{\pm 0.02}$ | $0.95_{\pm 0.02}$ |

Table 51: Overall results for mujoco unconstrained tasks with 128 solutions and 50th percentile evaluations. In this case, 20% of the values are missing near the worst value and another 30% near the optimal value. Details are the same as Table 5.

| Steps | | t = 50 | | | | | | t = 100 | | | | | | t = 150 | | |
|---|---|---|---|---|---|---|---|---|---|---|---|---|---|---|---|---|
| Task $f(x^*_{OFF})$ | | | | | | | | Mujoco 1 $0.45_{\pm 0.00}$ | | | | | | | | |
| Metric | FS | SI | OI | SO | SO$_\omega$ | FS | SI | OI | SO | SO$_\omega$ | FS | SI | OI | SO | SO$_\omega$ |
| ARCOO | $0.44_{\pm 0.00}$ | $0.99_{\pm 0.00}$ | $0.99_{\pm 0.00}$ | $0.99_{\pm 0.00}$ | $0.99_{\pm 0.00}$ | $0.44_{\pm 0.00}$ | $0.99_{\pm 0.00}$ | $0.99_{\pm 0.00}$ | $0.99_{\pm 0.00}$ | $0.99_{\pm 0.00}$ | $0.44_{\pm 0.00}$ | $0.99_{\pm 0.00}$ | $0.99_{\pm 0.00}$ | $0.99_{\pm 0.00}$ | $0.99_{\pm 0.00}$ |
| BO | $0.37_{\pm 0.03}$ | $0.82_{\pm 0.09}$ | $0.86_{\pm 0.05}$ | $0.83_{\pm 0.05}$ | $0.84_{\pm 0.03}$ | $0.36_{\pm 0.02}$ | $0.81_{\pm 0.09}$ | $0.86_{\pm 0.05}$ | $0.83_{\pm 0.05}$ | $0.82_{\pm 0.05}$ | $0.36_{\pm 0.04}$ | $0.80_{\pm 0.06}$ | $0.86_{\pm 0.09}$ | $0.83_{\pm 0.05}$ | $0.80_{\pm 0.08}$ |
| CBAS | $0.00_{\pm 0.00}$ | $-43.12_{\pm 0.18}$ | $-43.12_{\pm 0.18}$ | $-43.12_{\pm 0.18}$ | $-43.12_{\pm 0.18}$ | $0.00_{\pm 0.00}$ | $-43.35_{\pm 0.18}$ | $-43.35_{\pm 0.18}$ | $-43.35_{\pm 0.18}$ | $-43.35_{\pm 0.18}$ | $0.00_{\pm 0.00}$ | $-43.42_{\pm 0.18}$ | $-43.42_{\pm 0.18}$ | $-43.42_{\pm 0.18}$ | $-43.42_{\pm 0.18}$ |
| CCDDEA | $0.65_{\pm 0.14}$ | $0.78_{\pm 0.05}$ | $1.45_{\pm 0.12}$ | $1.01_{\pm 0.06}$ | $1.12_{\pm 0.07}$ | $0.65_{\pm 0.14}$ | $0.78_{\pm 0.10}$ | $1.45_{\pm 0.17}$ | $1.01_{\pm 0.12}$ | $0.92_{\pm 0.11}$ | $0.65_{\pm 0.14}$ | $0.78_{\pm 0.12}$ | $1.45_{\pm 0.21}$ | $1.01_{\pm 0.15}$ | $0.78_{\pm 0.12}$ |
| CMAES | $0.73_{\pm 0.06}$ | $0.93_{\pm 0.02}$ | $1.45_{\pm 0.07}$ | $1.13_{\pm 0.02}$ | $1.22_{\pm 0.03}$ | - | - | - | - | - | - | - | - | - | - |
| TTDDEA | $0.27_{\pm 0.06}$ | $0.39_{\pm 0.19}$ | $0.39_{\pm 0.19}$ | $0.39_{\pm 0.19}$ | $0.39_{\pm 0.19}$ | $0.28_{\pm 0.06}$ | $0.44_{\pm 0.18}$ | $0.44_{\pm 0.18}$ | $0.44_{\pm 0.18}$ | $0.44_{\pm 0.18}$ | $0.28_{\pm 0.06}$ | $0.45_{\pm 0.19}$ | $0.45_{\pm 0.19}$ | $0.45_{\pm 0.19}$ | $0.45_{\pm 0.19}$ |
| Trimentoring | $0.66_{\pm 0.11}$ | $0.91_{\pm 0.03}$ | $1.28_{\pm 0.14}$ | $1.06_{\pm 0.04}$ | $1.12_{\pm 0.06}$ | $0.66_{\pm 0.10}$ | $0.94_{\pm 0.03}$ | $1.34_{\pm 0.16}$ | $1.10_{\pm 0.05}$ | $1.04_{\pm 0.03}$ | $0.64_{\pm 0.09}$ | $0.95_{\pm 0.03}$ | $1.36_{\pm 0.16}$ | $1.11_{\pm 0.05}$ | $0.95_{\pm 0.03}$ |
| Task $f(x^*_{OFF})$ | | | | | | | | Mujoco 2 $0.03_{\pm 0.01}$ | | | | | | | | |
| ARCOO | $0.02_{\pm 0.00}$ | $0.99_{\pm 0.01}$ | $0.99_{\pm 0.01}$ | $0.99_{\pm 0.01}$ | $0.99_{\pm 0.01}$ | $0.02_{\pm 0.00}$ | $0.99_{\pm 0.01}$ | $0.99_{\pm 0.01}$ | $0.99_{\pm 0.01}$ | $0.99_{\pm 0.01}$ | $0.02_{\pm 0.00}$ | $0.99_{\pm 0.01}$ | $0.99_{\pm 0.01}$ | $0.99_{\pm 0.01}$ | $0.99_{\pm 0.01}$ |
| BO | $-0.10_{\pm 0.06}$ | $0.84_{\pm 0.07}$ | $0.87_{\pm 0.03}$ | $0.85_{\pm 0.05}$ | $0.86_{\pm 0.04}$ | $-0.09_{\pm 0.04}$ | $0.82_{\pm 0.08}$ | $0.87_{\pm 0.03}$ | $0.84_{\pm 0.05}$ | $0.84_{\pm 0.06}$ | $-0.07_{\pm 0.04}$ | $0.82_{\pm 0.07}$ | $0.87_{\pm 0.03}$ | $0.84_{\pm 0.05}$ | $0.82_{\pm 0.07}$ |
| CBAS | $-1.86_{\pm 0.00}$ | $-185.68_{\pm 0.45}$ | $-185.68_{\pm 0.45}$ | $-185.68_{\pm 0.45}$ | $-185.68_{\pm 0.45}$ | $-1.86_{\pm 0.00}$ | $-186.65_{\pm 0.46}$ | $-186.65_{\pm 0.46}$ | $-186.65_{\pm 0.46}$ | $-186.65_{\pm 0.46}$ | $-1.86_{\pm 0.00}$ | $-186.97_{\pm 0.46}$ | $-186.97_{\pm 0.46}$ | $-186.97_{\pm 0.46}$ | $-186.97_{\pm 0.46}$ |
| CCDDEA | $0.19_{\pm 0.25}$ | $0.75_{\pm 0.05}$ | $1.35_{\pm 0.08}$ | $0.97_{\pm 0.06}$ | $1.07_{\pm 0.06}$ | $0.26_{\pm 0.14}$ | $0.74_{\pm 0.06}$ | $1.35_{\pm 0.08}$ | $0.95_{\pm 0.07}$ | $0.87_{\pm 0.07}$ | $0.26_{\pm 0.14}$ | $0.73_{\pm 0.08}$ | $1.34_{\pm 0.11}$ | $0.95_{\pm 0.09}$ | $0.73_{\pm 0.08}$ |
| CMAES | $0.22_{\pm 0.04}$ | $0.97_{\pm 0.01}$ | $1.16_{\pm 0.04}$ | $1.05_{\pm 0.01}$ | $1.09_{\pm 0.02}$ | - | - | - | - | - | - | - | - | - | - |
| TTDDEA | $-0.35_{\pm 0.12}$ | $0.12_{\pm 0.24}$ | $0.12_{\pm 0.24}$ | $0.12_{\pm 0.24}$ | $0.12_{\pm 0.24}$ | $-0.29_{\pm 0.13}$ | $0.09_{\pm 0.26}$ | $0.09_{\pm 0.26}$ | $0.09_{\pm 0.26}$ | $0.09_{\pm 0.26}$ | $-0.29_{\pm 0.13}$ | $0.11_{\pm 0.28}$ | $0.11_{\pm 0.28}$ | $0.11_{\pm 0.28}$ | $0.11_{\pm 0.28}$ |
| Trimentoring | $0.19_{\pm 0.06}$ | $0.92_{\pm 0.02}$ | $1.14_{\pm 0.04}$ | $1.02_{\pm 0.03}$ | $1.05_{\pm 0.03}$ | $0.20_{\pm 0.07}$ | $0.93_{\pm 0.03}$ | $1.17_{\pm 0.06}$ | $1.03_{\pm 0.04}$ | $1.00_{\pm 0.04}$ | $0.18_{\pm 0.07}$ | $0.94_{\pm 0.03}$ | $1.18_{\pm 0.07}$ | $1.04_{\pm 0.05}$ | $0.94_{\pm 0.03}$ |

Table 52: Overall results for mujoco unconstrained tasks with 128 solutions and 50th percentile evaluations. In this case, 25% of the values are missing near the worst value and another 25% near the optimal value. Details are the same as Table 5.

| Steps | t = 50 | | | | | t = 100 | | | | | t = 150 | | | | |
|---|---|---|---|---|---|---|---|---|---|---|---|---|---|---|---|
| Task $f(\boldsymbol{x}_{OFF}^*)$ | | | | | | | | Mujoco 1 $0.47_{\pm0.00}$ | | | | | | | |
| Metric | FS | SI | OI | SO | $SO_\omega$ | FS | SI | OI | SO | $SO_\omega$ | FS | SI | OI | SO | $SO_\omega$ |
| ARCOO | $0.47_{\pm0.00}$ | $0.99_{\pm0.00}$ | $0.99_{\pm0.00}$ | $0.99_{\pm0.00}$ | $0.99_{\pm0.00}$ | $0.47_{\pm0.00}$ | $0.99_{\pm0.00}$ | $0.99_{\pm0.00}$ | $0.99_{\pm0.00}$ | $0.99_{\pm0.00}$ | $0.47_{\pm0.00}$ | $0.99_{\pm0.00}$ | $0.99_{\pm0.00}$ | $0.99_{\pm0.00}$ | $0.99_{\pm0.00}$ |
| BO | $0.38_{\pm0.04}$ | $0.77_{\pm0.09}$ | $0.80_{\pm0.01}$ | $0.78_{\pm0.05}$ | $0.79_{\pm0.04}$ | $0.36_{\pm0.03}$ | $0.77_{\pm0.09}$ | $0.80_{\pm0.01}$ | $0.78_{\pm0.05}$ | $0.78_{\pm0.07}$ | $0.37_{\pm0.04}$ | $0.77_{\pm0.09}$ | $0.80_{\pm0.01}$ | $0.78_{\pm0.05}$ | $0.77_{\pm0.09}$ |
| CBAS | $0.00_{\pm0.00}$ | $-45.62_{\pm0.25}$ | $-45.62_{\pm0.25}$ | $-45.62_{\pm0.25}$ | $-45.62_{\pm0.25}$ | $0.00_{\pm0.00}$ | $-45.86_{\pm0.26}$ | $-45.86_{\pm0.26}$ | $-45.86_{\pm0.26}$ | $-45.86_{\pm0.26}$ | $0.00_{\pm0.00}$ | $-45.94_{\pm0.26}$ | $-45.94_{\pm0.26}$ | $-45.94_{\pm0.26}$ | $-45.94_{\pm0.26}$ |
| CCDDEA | $0.69_{\pm0.17}$ | $0.69_{\pm0.08}$ | $1.33_{\pm0.20}$ | $0.91_{\pm0.11}$ | $1.02_{\pm0.13}$ | $0.73_{\pm0.13}$ | $0.76_{\pm0.09}$ | $1.48_{\pm0.21}$ | $1.01_{\pm0.12}$ | $0.91_{\pm0.11}$ | $0.73_{\pm0.13}$ | $0.79_{\pm0.10}$ | $1.52_{\pm0.23}$ | $1.04_{\pm0.14}$ | $0.79_{\pm0.10}$ |
| CMAES | $0.74_{\pm0.08}$ | $0.91_{\pm0.02}$ | $1.40_{\pm0.10}$ | $1.10_{\pm0.03}$ | $1.18_{\pm0.05}$ | - | - | - | - | - | - | - | - | - | - |
| TTDDEA | $0.28_{\pm0.08}$ | $0.05_{\pm0.46}$ | $0.05_{\pm0.46}$ | $0.05_{\pm0.46}$ | $0.05_{\pm0.46}$ | $0.29_{\pm0.09}$ | $0.10_{\pm0.42}$ | $0.10_{\pm0.42}$ | $0.10_{\pm0.42}$ | $0.10_{\pm0.42}$ | $0.29_{\pm0.07}$ | $0.12_{\pm0.42}$ | $0.12_{\pm0.42}$ | $0.12_{\pm0.42}$ | $0.12_{\pm0.42}$ |
| Trimentoring | $0.65_{\pm0.08}$ | $0.91_{\pm0.03}$ | $1.24_{\pm0.11}$ | $1.04_{\pm0.04}$ | $1.10_{\pm0.06}$ | $0.65_{\pm0.09}$ | $0.93_{\pm0.03}$ | $1.30_{\pm0.14}$ | $1.08_{\pm0.05}$ | $1.02_{\pm0.03}$ | $0.65_{\pm0.08}$ | $0.94_{\pm0.03}$ | $1.32_{\pm0.15}$ | $1.09_{\pm0.06}$ | $0.94_{\pm0.03}$ |
| Task $f(\boldsymbol{x}_{OFF}^*)$ | | | | | | | | Mujoco 2 $0.07_{\pm0.00}$ | | | | | | | |
| ARCOO | $0.07_{\pm0.00}$ | $0.99_{\pm0.00}$ | $0.99_{\pm0.00}$ | $0.99_{\pm0.00}$ | $0.99_{\pm0.00}$ | $0.07_{\pm0.00}$ | $0.99_{\pm0.00}$ | $0.99_{\pm0.00}$ | $0.99_{\pm0.00}$ | $0.99_{\pm0.00}$ | $0.07_{\pm0.00}$ | $0.99_{\pm0.01}$ | $0.99_{\pm0.01}$ | $0.99_{\pm0.01}$ | $0.99_{\pm0.01}$ |
| BO | $-0.09_{\pm0.08}$ | $0.85_{\pm0.05}$ | $0.85_{\pm0.05}$ | $0.85_{\pm0.05}$ | $0.85_{\pm0.05}$ | $-0.13_{\pm0.04}$ | $0.85_{\pm0.05}$ | $0.85_{\pm0.05}$ | $0.85_{\pm0.05}$ | $0.85_{\pm0.05}$ | $-0.09_{\pm0.05}$ | $0.85_{\pm0.05}$ | $0.85_{\pm0.05}$ | $0.85_{\pm0.05}$ | $0.85_{\pm0.05}$ |
| CBAS | $-1.86_{\pm0.00}$ | $-189.94_{\pm0.32}$ | $-189.94_{\pm0.32}$ | $-189.94_{\pm0.32}$ | $-189.94_{\pm0.32}$ | $-1.86_{\pm0.00}$ | $-190.93_{\pm0.33}$ | $-190.93_{\pm0.33}$ | $-190.93_{\pm0.33}$ | $-190.93_{\pm0.33}$ | $-1.86_{\pm0.00}$ | $-191.26_{\pm0.33}$ | $-191.26_{\pm0.33}$ | $-191.26_{\pm0.33}$ | $-191.26_{\pm0.33}$ |
| CCDDEA | $0.08_{\pm0.25}$ | $0.73_{\pm0.06}$ | $1.34_{\pm0.14}$ | $0.95_{\pm0.07}$ | $1.05_{\pm0.09}$ | $0.33_{\pm0.18}$ | $0.73_{\pm0.09}$ | $1.35_{\pm0.19}$ | $0.95_{\pm0.11}$ | $0.87_{\pm0.10}$ | $0.33_{\pm0.18}$ | $0.75_{\pm0.11}$ | $1.37_{\pm0.23}$ | $0.97_{\pm0.14}$ | $0.75_{\pm0.11}$ |
| CMAES | $0.22_{\pm0.03}$ | $0.97_{\pm0.01}$ | $1.12_{\pm0.03}$ | $1.04_{\pm0.01}$ | $1.07_{\pm0.01}$ | - | - | - | - | - | - | - | - | - | - |
| TTDDEA | $-0.25_{\pm0.13}$ | $0.16_{\pm0.31}$ | $0.16_{\pm0.31}$ | $0.16_{\pm0.31}$ | $0.16_{\pm0.31}$ | $-0.29_{\pm0.17}$ | $0.11_{\pm0.38}$ | $0.11_{\pm0.38}$ | $0.11_{\pm0.38}$ | $0.11_{\pm0.38}$ | $-0.31_{\pm0.14}$ | $0.09_{\pm0.39}$ | $0.09_{\pm0.39}$ | $0.09_{\pm0.39}$ | $0.09_{\pm0.39}$ |
| Trimentoring | $0.28_{\pm0.11}$ | $0.92_{\pm0.03}$ | $1.14_{\pm0.08}$ | $1.02_{\pm0.03}$ | $1.06_{\pm0.04}$ | $0.25_{\pm0.11}$ | $0.94_{\pm0.03}$ | $1.18_{\pm0.10}$ | $1.05_{\pm0.05}$ | $1.01_{\pm0.04}$ | $0.24_{\pm0.11}$ | $0.95_{\pm0.03}$ | $1.18_{\pm0.10}$ | $1.05_{\pm0.05}$ | $0.95_{\pm0.03}$ |

Table 53: Overall results for mujoco unconstrained tasks with 128 solutions and 50th percentile evaluations. In this case, 30% of the values are missing near the worst value and another 20% near the optimal value. Details are the same as Table 5.

| Steps | t = 50 | | | | | t = 100 | | | | | t = 150 | | | | |
|---|---|---|---|---|---|---|---|---|---|---|---|---|---|---|---|
| Task $f(\boldsymbol{x}_{OFF}^*)$ | | | | | | | | Mujoco 1 $0.50_{\pm0.00}$ | | | | | | | |
| Metric | FS | SI | OI | SO | $SO_\omega$ | FS | SI | OI | SO | $SO_\omega$ | FS | SI | OI | SO | $SO_\omega$ |
| ARCOO | $0.50_{\pm0.00}$ | $0.99_{\pm0.00}$ | $0.99_{\pm0.00}$ | $0.99_{\pm0.00}$ | $0.99_{\pm0.00}$ | $0.49_{\pm0.00}$ | $0.99_{\pm0.00}$ | $0.99_{\pm0.00}$ | $0.99_{\pm0.00}$ | $0.99_{\pm0.00}$ | $0.49_{\pm0.00}$ | $0.98_{\pm0.00}$ | $0.98_{\pm0.00}$ | $0.98_{\pm0.00}$ | $0.98_{\pm0.00}$ |
| BO | $0.37_{\pm0.03}$ | $0.67_{\pm0.09}$ | $0.73_{\pm0.01}$ | $0.69_{\pm0.05}$ | $0.70_{\pm0.04}$ | $0.36_{\pm0.02}$ | $0.67_{\pm0.09}$ | $0.72_{\pm0.01}$ | $0.69_{\pm0.06}$ | $0.68_{\pm0.07}$ | $0.39_{\pm0.04}$ | $0.67_{\pm0.09}$ | $0.72_{\pm0.01}$ | $0.69_{\pm0.06}$ | $0.67_{\pm0.09}$ |
| CBAS | $0.00_{\pm0.00}$ | $-48.64_{\pm0.34}$ | $-48.64_{\pm0.34}$ | $-48.64_{\pm0.34}$ | $-48.64_{\pm0.34}$ | $0.00_{\pm0.00}$ | $-48.89_{\pm0.34}$ | $-48.89_{\pm0.34}$ | $-48.89_{\pm0.34}$ | $-48.89_{\pm0.34}$ | $0.00_{\pm0.00}$ | $-48.98_{\pm0.34}$ | $-48.98_{\pm0.34}$ | $-48.98_{\pm0.34}$ | $-48.98_{\pm0.34}$ |
| CCDDEA | $0.73_{\pm0.14}$ | $0.70_{\pm0.06}$ | $1.35_{\pm0.14}$ | $0.92_{\pm0.09}$ | $1.03_{\pm0.10}$ | $0.73_{\pm0.14}$ | $0.77_{\pm0.12}$ | $1.48_{\pm0.23}$ | $1.01_{\pm0.16}$ | $0.92_{\pm0.14}$ | $0.73_{\pm0.14}$ | $0.79_{\pm0.14}$ | $1.53_{\pm0.27}$ | $1.04_{\pm0.19}$ | $0.79_{\pm0.15}$ |
| CMAES | $0.75_{\pm0.05}$ | $0.92_{\pm0.01}$ | $1.40_{\pm0.08}$ | $1.11_{\pm0.03}$ | $1.19_{\pm0.04}$ | - | - | - | - | - | - | - | - | - | - |
| TTDDEA | $0.28_{\pm0.09}$ | $-0.36_{\pm0.54}$ | $-0.36_{\pm0.54}$ | $-0.36_{\pm0.54}$ | $-0.36_{\pm0.54}$ | $0.32_{\pm0.09}$ | $-0.23_{\pm0.46}$ | $-0.23_{\pm0.46}$ | $-0.23_{\pm0.46}$ | $-0.23_{\pm0.46}$ | $0.32_{\pm0.09}$ | $-0.18_{\pm0.44}$ | $-0.18_{\pm0.44}$ | $-0.18_{\pm0.44}$ | $-0.18_{\pm0.44}$ |
| Trimentoring | $0.64_{\pm0.09}$ | $0.90_{\pm0.03}$ | $1.19_{\pm0.13}$ | $1.02_{\pm0.05}$ | $1.07_{\pm0.07}$ | $0.63_{\pm0.10}$ | $0.92_{\pm0.02}$ | $1.23_{\pm0.16}$ | $1.05_{\pm0.07}$ | $1.00_{\pm0.05}$ | $0.63_{\pm0.10}$ | $0.93_{\pm0.03}$ | $1.24_{\pm0.18}$ | $1.06_{\pm0.08}$ | $0.93_{\pm0.03}$ |
| Task $f(\boldsymbol{x}_{OFF}^*)$ | | | | | | | | Mujoco 2 $0.11_{\pm0.00}$ | | | | | | | |
| ARCOO | $0.11_{\pm0.00}$ | $0.99_{\pm0.00}$ | $0.99_{\pm0.00}$ | $0.99_{\pm0.00}$ | $0.99_{\pm0.00}$ | $0.11_{\pm0.00}$ | $0.99_{\pm0.00}$ | $0.99_{\pm0.00}$ | $0.99_{\pm0.00}$ | $0.99_{\pm0.00}$ | $0.11_{\pm0.00}$ | $0.99_{\pm0.00}$ | $0.99_{\pm0.00}$ | $0.99_{\pm0.00}$ | $0.99_{\pm0.00}$ |
| BO | $-0.12_{\pm0.05}$ | $0.80_{\pm0.05}$ | $0.80_{\pm0.05}$ | $0.80_{\pm0.05}$ | $0.80_{\pm0.05}$ | $-0.10_{\pm0.03}$ | $0.80_{\pm0.06}$ | $0.80_{\pm0.06}$ | $0.80_{\pm0.06}$ | $0.80_{\pm0.06}$ | $-0.11_{\pm0.03}$ | $0.80_{\pm0.05}$ | $0.80_{\pm0.05}$ | $0.80_{\pm0.05}$ | $0.80_{\pm0.05}$ |
| CBAS | $-1.86_{\pm0.00}$ | $-194.21_{\pm0.35}$ | $-194.21_{\pm0.35}$ | $-194.21_{\pm0.35}$ | $-194.21_{\pm0.35}$ | $-1.86_{\pm0.00}$ | $-195.23_{\pm0.35}$ | $-195.23_{\pm0.35}$ | $-195.23_{\pm0.35}$ | $-195.23_{\pm0.35}$ | $-1.86_{\pm0.00}$ | $-195.56_{\pm0.35}$ | $-195.56_{\pm0.35}$ | $-195.56_{\pm0.35}$ | $-195.56_{\pm0.35}$ |
| CCDDEA | $0.18_{\pm0.21}$ | $0.69_{\pm0.07}$ | $1.24_{\pm0.14}$ | $0.89_{\pm0.09}$ | $0.98_{\pm0.10}$ | $0.29_{\pm0.11}$ | $0.67_{\pm0.05}$ | $1.24_{\pm0.07}$ | $0.87_{\pm0.04}$ | $0.80_{\pm0.04}$ | $0.29_{\pm0.11}$ | $0.67_{\pm0.06}$ | $1.24_{\pm0.07}$ | $0.87_{\pm0.06}$ | $0.68_{\pm0.06}$ |
| CMAES | $0.24_{\pm0.02}$ | $0.97_{\pm0.01}$ | $1.10_{\pm0.01}$ | $1.03_{\pm0.01}$ | $1.05_{\pm0.01}$ | - | - | - | - | - | - | - | - | - | - |
| TTDDEA | $-0.20_{\pm0.14}$ | $0.04_{\pm0.33}$ | $0.04_{\pm0.33}$ | $0.04_{\pm0.33}$ | $0.04_{\pm0.33}$ | $-0.25_{\pm0.15}$ | $0.03_{\pm0.36}$ | $0.03_{\pm0.36}$ | $0.03_{\pm0.36}$ | $0.03_{\pm0.36}$ | $-0.25_{\pm0.15}$ | $0.03_{\pm0.37}$ | $0.03_{\pm0.37}$ | $0.03_{\pm0.37}$ | $0.03_{\pm0.37}$ |
| Trimentoring | $0.20_{\pm0.07}$ | $0.93_{\pm0.05}$ | $1.06_{\pm0.04}$ | $0.99_{\pm0.04}$ | $1.01_{\pm0.03}$ | $0.23_{\pm0.07}$ | $0.94_{\pm0.06}$ | $1.08_{\pm0.05}$ | $1.00_{\pm0.05}$ | $0.98_{\pm0.05}$ | $0.23_{\pm0.07}$ | $0.94_{\pm0.06}$ | $1.09_{\pm0.06}$ | $1.01_{\pm0.06}$ | $0.94_{\pm0.06}$ |

Table 54: Overall results for mujoco unconstrained tasks with 128 solutions and 50th percentile evaluations. In this case, 40% of the values are missing near the worst value and another 10% near the optimal value. Details are the same as Table 5.

| Steps | t = 50 | | | | | t = 100 | | | | | t = 150 | | | | |
|---|---|---|---|---|---|---|---|---|---|---|---|---|---|---|---|
| Task $f(\boldsymbol{x}_{OFF}^*)$ | | | | | | | | Mujoco 1 $0.59_{\pm0.01}$ | | | | | | | |
| Metric | FS | SI | OI | SO | $SO_\omega$ | FS | SI | OI | SO | $SO_\omega$ | FS | SI | OI | SO | $SO_\omega$ |
| ARCOO | $0.58_{\pm0.01}$ | $0.97_{\pm0.01}$ | $0.97_{\pm0.01}$ | $0.97_{\pm0.01}$ | $0.97_{\pm0.01}$ | $0.58_{\pm0.01}$ | $0.97_{\pm0.00}$ | $0.97_{\pm0.00}$ | $0.97_{\pm0.00}$ | $0.97_{\pm0.00}$ | $0.58_{\pm0.01}$ | $0.96_{\pm0.01}$ | $0.96_{\pm0.01}$ | $0.96_{\pm0.01}$ | $0.96_{\pm0.01}$ |
| BO | $0.38_{\pm0.01}$ | $0.43_{\pm0.04}$ | $0.43_{\pm0.04}$ | $0.43_{\pm0.04}$ | $0.43_{\pm0.04}$ | $0.36_{\pm0.02}$ | $0.42_{\pm0.05}$ | $0.43_{\pm0.03}$ | $0.42_{\pm0.04}$ | $0.42_{\pm0.04}$ | $0.36_{\pm0.02}$ | $0.41_{\pm0.04}$ | $0.43_{\pm0.03}$ | $0.42_{\pm0.03}$ | $0.41_{\pm0.04}$ |
| CBAS | $0.00_{\pm0.00}$ | $-57.79_{\pm0.58}$ | $-57.79_{\pm0.58}$ | $-57.79_{\pm0.58}$ | $-57.79_{\pm0.58}$ | $0.00_{\pm0.00}$ | $-58.10_{\pm0.58}$ | $-58.10_{\pm0.58}$ | $-58.10_{\pm0.58}$ | $-58.10_{\pm0.58}$ | $0.00_{\pm0.00}$ | $-58.20_{\pm0.59}$ | $-58.20_{\pm0.59}$ | $-58.20_{\pm0.59}$ | $-58.20_{\pm0.59}$ |
| CCDDEA | $0.74_{\pm0.11}$ | $0.62_{\pm0.17}$ | $1.18_{\pm0.33}$ | $0.81_{\pm0.23}$ | $0.91_{\pm0.25}$ | $0.76_{\pm0.06}$ | $0.71_{\pm0.12}$ | $1.36_{\pm0.19}$ | $0.93_{\pm0.14}$ | $0.85_{\pm0.13}$ | $0.76_{\pm0.06}$ | $0.74_{\pm0.11}$ | $1.42_{\pm0.17}$ | $0.97_{\pm0.13}$ | $0.74_{\pm0.11}$ |
| CMAES | $0.79_{\pm0.05}$ | $0.92_{\pm0.02}$ | $1.39_{\pm0.06}$ | $1.10_{\pm0.01}$ | $1.19_{\pm0.02}$ | - | - | - | - | - | - | - | - | - | - |
| TTDDEA | $0.16_{\pm0.08}$ | $-5.61_{\pm6.45}$ | $-5.61_{\pm6.45}$ | $-5.61_{\pm6.45}$ | $-5.61_{\pm6.45}$ | $0.27_{\pm0.10}$ | $-5.45_{\pm6.18}$ | $-5.45_{\pm6.18}$ | $-5.45_{\pm6.18}$ | $-5.45_{\pm6.18}$ | $0.22_{\pm0.10}$ | $-5.08_{\pm5.81}$ | $-5.08_{\pm5.81}$ | $-5.08_{\pm5.81}$ | $-5.08_{\pm5.81}$ |
| Trimentoring | $0.71_{\pm0.10}$ | $0.88_{\pm0.05}$ | $1.18_{\pm0.18}$ | $1.00_{\pm0.09}$ | $1.05_{\pm0.12}$ | $0.71_{\pm0.11}$ | $0.89_{\pm0.07}$ | $1.22_{\pm0.24}$ | $1.03_{\pm0.13}$ | $0.98_{\pm0.11}$ | $0.71_{\pm0.11}$ | $0.90_{\pm0.08}$ | $1.25_{\pm0.25}$ | $1.04_{\pm0.14}$ | $0.91_{\pm0.08}$ |
| Task $f(\boldsymbol{x}_{OFF}^*)$ | | | | | | | | Mujoco 1 $0.23_{\pm0.01}$ | | | | | | | |
| ARCOO | $0.22_{\pm0.01}$ | $0.97_{\pm0.01}$ | $0.97_{\pm0.01}$ | $0.97_{\pm0.01}$ | $0.97_{\pm0.01}$ | $0.22_{\pm0.01}$ | $0.97_{\pm0.01}$ | $0.97_{\pm0.01}$ | $0.97_{\pm0.01}$ | $0.97_{\pm0.01}$ | $0.22_{\pm0.01}$ | $0.97_{\pm0.02}$ | $0.97_{\pm0.02}$ | $0.97_{\pm0.02}$ | $0.97_{\pm0.02}$ |
| BO | $-0.10_{\pm0.05}$ | $0.67_{\pm0.12}$ | $0.67_{\pm0.12}$ | $0.67_{\pm0.12}$ | $0.67_{\pm0.12}$ | $-0.10_{\pm0.05}$ | $0.67_{\pm0.12}$ | $0.67_{\pm0.12}$ | $0.67_{\pm0.12}$ | $0.67_{\pm0.12}$ | $-0.10_{\pm0.03}$ | $0.67_{\pm0.12}$ | $0.67_{\pm0.12}$ | $0.67_{\pm0.12}$ | $0.67_{\pm0.12}$ |
| CBAS | $-1.86_{\pm0.00}$ | $-205.86_{\pm0.46}$ | $-205.86_{\pm0.46}$ | $-205.86_{\pm0.46}$ | $-205.86_{\pm0.46}$ | $-1.86_{\pm0.00}$ | $-206.94_{\pm0.46}$ | $-206.94_{\pm0.46}$ | $-206.94_{\pm0.46}$ | $-206.94_{\pm0.46}$ | $-1.86_{\pm0.00}$ | $-207.29_{\pm0.46}$ | $-207.29_{\pm0.46}$ | $-207.29_{\pm0.46}$ | $-207.29_{\pm0.46}$ |
| CCDDEA | $0.17_{\pm0.20}$ | $0.61_{\pm0.07}$ | $1.10_{\pm0.13}$ | $0.78_{\pm0.08}$ | $0.86_{\pm0.08}$ | $0.27_{\pm0.13}$ | $0.58_{\pm0.07}$ | $1.08_{\pm0.14}$ | $0.76_{\pm0.08}$ | $0.69_{\pm0.08}$ | $0.27_{\pm0.13}$ | $0.58_{\pm0.09}$ | $1.06_{\pm0.17}$ | $0.75_{\pm0.11}$ | $0.58_{\pm0.09}$ |
| CMAES | $0.25_{\pm0.03}$ | $0.95_{\pm0.01}$ | $1.00_{\pm0.02}$ | $0.98_{\pm0.01}$ | $0.98_{\pm0.02}$ | - | - | - | - | - | - | - | - | - | - |
| TTDDEA | $-0.27_{\pm0.02}$ | $-8.29_{\pm14.68}$ | $-8.29_{\pm14.68}$ | $-8.29_{\pm14.68}$ | $-8.29_{\pm14.68}$ | $-0.27_{\pm0.01}$ | $-8.41_{\pm14.91}$ | $-8.41_{\pm14.91}$ | $-8.41_{\pm14.91}$ | $-8.41_{\pm14.91}$ | $-0.27_{\pm0.01}$ | $-8.51_{\pm15.14}$ | $-8.51_{\pm15.14}$ | $-8.51_{\pm15.14}$ | $-8.51_{\pm15.14}$ |
| Trimentoring | $0.25_{\pm0.11}$ | $0.90_{\pm0.07}$ | $0.98_{\pm0.10}$ | $0.93_{\pm0.09}$ | $0.95_{\pm0.09}$ | $0.26_{\pm0.09}$ | $0.90_{\pm0.07}$ | $0.99_{\pm0.10}$ | $0.94_{\pm0.08}$ | $0.93_{\pm0.08}$ | $0.25_{\pm0.09}$ | $0.91_{\pm0.07}$ | $1.00_{\pm0.10}$ | $0.95_{\pm0.08}$ | $0.91_{\pm0.07}$ |

Table 55: Overall results for mujoco unconstrained tasks with 128 solutions and 0th percentile evaluations. In this case, 0% of the values are missing near the worst value and another 50% near the optimal value. Details are the same as Table 5.

| Steps | | t = 50 | | | | | t = 100 | | | | | t = 150 | | | |
|---|---|---|---|---|---|---|---|---|---|---|---|---|---|---|---|
| Task $f(x^*_{OFF})$ | | | | | | | Mujoco 1 $0.37_{\pm0.00}$ | | | | | | | | |
| Metric | FS | SI | OI | SO | SO$_\omega$ | FS | SI | OI | SO | SO$_\omega$ | FS | SI | OI | SO | SO$_\omega$ |
| ARCOO | $0.35_{\pm0.03}$ | $0.98_{\pm0.02}$ | $0.98_{\pm0.02}$ | $0.98_{\pm0.02}$ | $0.98_{\pm0.02}$ | $0.15_{\pm0.05}$ | $0.77_{\pm0.09}$ | $0.77_{\pm0.09}$ | $0.77_{\pm0.09}$ | $0.77_{\pm0.09}$ | $0.14_{\pm0.04}$ | $0.63_{\pm0.12}$ | $0.63_{\pm0.12}$ | $0.63_{\pm0.12}$ | $0.63_{\pm0.12}$ |
| BO | $0.14_{\pm0.02}$ | $0.64_{\pm0.01}$ | $0.64_{\pm0.01}$ | $0.64_{\pm0.01}$ | $0.64_{\pm0.01}$ | $0.14_{\pm0.06}$ | $0.64_{\pm0.01}$ | $0.64_{\pm0.01}$ | $0.64_{\pm0.01}$ | $0.64_{\pm0.01}$ | $0.16_{\pm0.07}$ | $0.64_{\pm0.01}$ | $0.64_{\pm0.01}$ | $0.64_{\pm0.01}$ | $0.64_{\pm0.01}$ |
| CBAS | $0.00_{\pm0.00}$ | $-35.22_{\pm0.12}$ | $-35.22_{\pm0.12}$ | $-35.22_{\pm0.12}$ | $-35.22_{\pm0.12}$ | $0.00_{\pm0.00}$ | $-35.41_{\pm0.12}$ | $-35.41_{\pm0.12}$ | $-35.41_{\pm0.12}$ | $-35.41_{\pm0.12}$ | $0.00_{\pm0.00}$ | $-35.47_{\pm0.12}$ | $-35.47_{\pm0.12}$ | $-35.47_{\pm0.12}$ | $-35.47_{\pm0.12}$ |
| CCDDEA | $0.75_{\pm0.09}$ | $0.74_{\pm0.03}$ | $1.41_{\pm0.08}$ | $0.97_{\pm0.02}$ | $1.09_{\pm0.03}$ | $0.78_{\pm0.07}$ | $0.83_{\pm0.03}$ | $1.58_{\pm0.10}$ | $1.09_{\pm0.03}$ | $0.99_{\pm0.03}$ | $0.78_{\pm0.07}$ | $0.86_{\pm0.04}$ | $1.64_{\pm0.11}$ | $1.13_{\pm0.04}$ | $0.87_{\pm0.04}$ |
| CMAES | $0.34_{\pm0.04}$ | $0.85_{\pm0.03}$ | $0.89_{\pm0.05}$ | $0.87_{\pm0.04}$ | $0.87_{\pm0.04}$ | - | - | - | - | - | - | - | - | - | - |
| TTDDEA | $0.19_{\pm0.11}$ | $0.40_{\pm0.18}$ | $0.42_{\pm0.20}$ | $0.41_{\pm0.19}$ | $0.41_{\pm0.20}$ | $0.19_{\pm0.11}$ | $0.41_{\pm0.23}$ | $0.43_{\pm0.26}$ | $0.42_{\pm0.25}$ | $0.42_{\pm0.24}$ | $0.22_{\pm0.09}$ | $0.43_{\pm0.21}$ | $0.45_{\pm0.24}$ | $0.44_{\pm0.22}$ | $0.43_{\pm0.21}$ |
| Trimentoring | $0.26_{\pm0.06}$ | $0.79_{\pm0.04}$ | $0.80_{\pm0.05}$ | $0.80_{\pm0.04}$ | $0.80_{\pm0.04}$ | $0.30_{\pm0.07}$ | $0.82_{\pm0.06}$ | $0.84_{\pm0.08}$ | $0.83_{\pm0.07}$ | $0.83_{\pm0.06}$ | $0.34_{\pm0.06}$ | $0.84_{\pm0.05}$ | $0.87_{\pm0.09}$ | $0.85_{\pm0.06}$ | $0.84_{\pm0.05}$ |
| Task $f(x^*_{OFF})$ | | | | | | | Mujoco 2 $-0.1_{\pm0.00}$ | | | | | | | | |
| ARCOO | $-0.22_{\pm0.17}$ | $0.98_{\pm0.01}$ | $0.98_{\pm0.01}$ | $0.98_{\pm0.01}$ | $0.98_{\pm0.01}$ | $-0.75_{\pm0.34}$ | $0.69_{\pm0.12}$ | $0.69_{\pm0.12}$ | $0.69_{\pm0.12}$ | $0.69_{\pm0.12}$ | $-0.94_{\pm0.24}$ | $0.37_{\pm0.24}$ | $0.37_{\pm0.24}$ | $0.37_{\pm0.24}$ | $0.37_{\pm0.24}$ |
| BO | $-0.99_{\pm0.53}$ | $0.42_{\pm0.17}$ | $0.42_{\pm0.18}$ | $0.42_{\pm0.18}$ | $0.42_{\pm0.18}$ | $-1.07_{\pm0.51}$ | $0.40_{\pm0.19}$ | $0.40_{\pm0.19}$ | $0.40_{\pm0.19}$ | $0.40_{\pm0.19}$ | $-0.74_{\pm0.34}$ | $0.40_{\pm0.19}$ | $0.40_{\pm0.19}$ | $0.40_{\pm0.19}$ | $0.40_{\pm0.19}$ |
| CBAS | $-1.86_{\pm0.00}$ | $-173.04_{\pm0.27}$ | $-173.04_{\pm0.27}$ | $-173.04_{\pm0.27}$ | $-173.04_{\pm0.27}$ | $-1.86_{\pm0.00}$ | $-173.95_{\pm0.27}$ | $-173.95_{\pm0.27}$ | $-173.95_{\pm0.27}$ | $-173.95_{\pm0.27}$ | $-1.86_{\pm0.00}$ | $-174.25_{\pm0.27}$ | $-174.25_{\pm0.27}$ | $-174.25_{\pm0.27}$ | $-174.25_{\pm0.27}$ |
| CCDDEA | $0.06_{\pm0.29}$ | $0.72_{\pm0.08}$ | $1.27_{\pm0.10}$ | $0.92_{\pm0.09}$ | $1.01_{\pm0.09}$ | $0.25_{\pm0.11}$ | $0.77_{\pm0.07}$ | $1.38_{\pm0.11}$ | $0.98_{\pm0.07}$ | $0.90_{\pm0.07}$ | $0.25_{\pm0.11}$ | $0.80_{\pm0.07}$ | $1.44_{\pm0.13}$ | $1.03_{\pm0.08}$ | $0.80_{\pm0.07}$ |
| CMAES | $-0.22_{\pm0.11}$ | $0.74_{\pm0.09}$ | $0.75_{\pm0.08}$ | $0.75_{\pm0.08}$ | $0.75_{\pm0.08}$ | - | - | - | - | - | - | - | - | - | - |
| TTDDEA | $-0.74_{\pm0.77}$ | $0.46_{\pm0.32}$ | $0.52_{\pm0.37}$ | $0.49_{\pm0.34}$ | $0.50_{\pm0.35}$ | $-0.67_{\pm0.58}$ | $0.52_{\pm0.31}$ | $0.58_{\pm0.36}$ | $0.55_{\pm0.33}$ | $0.54_{\pm0.32}$ | $-0.66_{\pm0.55}$ | $0.54_{\pm0.32}$ | $0.62_{\pm0.37}$ | $0.58_{\pm0.34}$ | $0.54_{\pm0.32}$ |
| Trimentoring | $-0.07_{\pm0.17}$ | $0.68_{\pm0.10}$ | $0.80_{\pm0.12}$ | $0.73_{\pm0.10}$ | $0.75_{\pm0.10}$ | $-0.04_{\pm0.09}$ | $0.80_{\pm0.06}$ | $0.95_{\pm0.14}$ | $0.87_{\pm0.09}$ | $0.84_{\pm0.07}$ | $-0.05_{\pm0.10}$ | $0.84_{\pm0.03}$ | $0.99_{\pm0.12}$ | $0.91_{\pm0.06}$ | $0.84_{\pm0.03}$ |

Table 56: Overall results for mujoco unconstrained tasks with 128 solutions and 0th percentile evaluations. In this case, 0% of the values are missing near the worst value and another 40% near the optimal value. Details are the same as Table 5.

| Steps | | t = 50 | | | | | t = 100 | | | | | t = 150 | | | |
|---|---|---|---|---|---|---|---|---|---|---|---|---|---|---|---|
| Task $f(x^*_{OFF})$ | | | | | | | Mujoco 1 $0.4_{\pm0.00}$ | | | | | | | | |
| Metric | FS | SI | OI | SO | SO$_\omega$ | FS | SI | OI | SO | SO$_\omega$ | FS | SI | OI | SO | SO$_\omega$ |
| ARCOO | $0.40_{\pm0.00}$ | $0.98_{\pm0.01}$ | $0.98_{\pm0.01}$ | $0.98_{\pm0.01}$ | $0.98_{\pm0.01}$ | $0.15_{\pm0.03}$ | $0.82_{\pm0.06}$ | $0.82_{\pm0.06}$ | $0.82_{\pm0.06}$ | $0.82_{\pm0.06}$ | $0.14_{\pm0.03}$ | $0.64_{\pm0.15}$ | $0.64_{\pm0.15}$ | $0.64_{\pm0.15}$ | $0.64_{\pm0.15}$ |
| BO | $0.14_{\pm0.06}$ | $0.56_{\pm0.01}$ | $0.56_{\pm0.01}$ | $0.56_{\pm0.01}$ | $0.56_{\pm0.01}$ | $0.15_{\pm0.05}$ | $0.56_{\pm0.01}$ | $0.56_{\pm0.01}$ | $0.56_{\pm0.01}$ | $0.56_{\pm0.01}$ | $0.15_{\pm0.02}$ | $0.56_{\pm0.01}$ | $0.56_{\pm0.01}$ | $0.56_{\pm0.01}$ | $0.56_{\pm0.01}$ |
| CBAS | $0.00_{\pm0.00}$ | $-38.87_{\pm0.22}$ | $-38.87_{\pm0.22}$ | $-38.87_{\pm0.22}$ | $-38.87_{\pm0.22}$ | $0.00_{\pm0.00}$ | $-39.08_{\pm0.22}$ | $-39.08_{\pm0.22}$ | $-39.08_{\pm0.22}$ | $-39.08_{\pm0.22}$ | $0.00_{\pm0.00}$ | $-39.15_{\pm0.22}$ | $-39.15_{\pm0.22}$ | $-39.15_{\pm0.22}$ | $-39.15_{\pm0.22}$ |
| CCDDEA | $0.73_{\pm0.16}$ | $0.74_{\pm0.08}$ | $1.43_{\pm0.17}$ | $0.98_{\pm0.11}$ | $1.09_{\pm0.12}$ | $0.77_{\pm0.15}$ | $0.82_{\pm0.09}$ | $1.58_{\pm0.17}$ | $1.08_{\pm0.11}$ | $0.98_{\pm0.10}$ | $0.77_{\pm0.15}$ | $0.84_{\pm0.10}$ | $1.63_{\pm0.19}$ | $1.11_{\pm0.13}$ | $0.85_{\pm0.10}$ |
| CMAES | $0.36_{\pm0.07}$ | $0.80_{\pm0.04}$ | $0.82_{\pm0.05}$ | $0.81_{\pm0.05}$ | $0.81_{\pm0.05}$ | - | - | - | - | - | - | - | - | - | - |
| TTDDEA | $0.18_{\pm0.05}$ | $0.27_{\pm0.14}$ | $0.27_{\pm0.14}$ | $0.27_{\pm0.14}$ | $0.27_{\pm0.14}$ | $0.21_{\pm0.10}$ | $0.31_{\pm0.16}$ | $0.31_{\pm0.16}$ | $0.31_{\pm0.16}$ | $0.31_{\pm0.16}$ | $0.21_{\pm0.10}$ | $0.33_{\pm0.17}$ | $0.33_{\pm0.17}$ | $0.33_{\pm0.17}$ | $0.33_{\pm0.17}$ |
| Trimentoring | $0.27_{\pm0.06}$ | $0.65_{\pm0.14}$ | $0.65_{\pm0.14}$ | $0.65_{\pm0.14}$ | $0.65_{\pm0.14}$ | $0.31_{\pm0.06}$ | $0.69_{\pm0.16}$ | $0.69_{\pm0.16}$ | $0.69_{\pm0.16}$ | $0.69_{\pm0.16}$ | $0.34_{\pm0.05}$ | $0.72_{\pm0.17}$ | $0.73_{\pm0.17}$ | $0.72_{\pm0.17}$ | $0.72_{\pm0.17}$ |
| Task $f(x^*_{OFF})$ | | | | | | | Mujoco 2 $0.4_{\pm0.00}$ | | | | | | | | |
| ARCOO | $0.40_{\pm0.00}$ | $0.98_{\pm0.01}$ | $0.98_{\pm0.01}$ | $0.98_{\pm0.01}$ | $0.98_{\pm0.01}$ | $0.15_{\pm0.03}$ | $0.82_{\pm0.06}$ | $0.82_{\pm0.06}$ | $0.82_{\pm0.06}$ | $0.82_{\pm0.06}$ | $0.14_{\pm0.03}$ | $0.64_{\pm0.15}$ | $0.64_{\pm0.15}$ | $0.64_{\pm0.15}$ | $0.64_{\pm0.15}$ |
| BO | $-0.67_{\pm0.16}$ | $0.32_{\pm0.14}$ | $0.32_{\pm0.14}$ | $0.32_{\pm0.14}$ | $0.32_{\pm0.14}$ | $-0.84_{\pm0.32}$ | $0.30_{\pm0.15}$ | $0.30_{\pm0.15}$ | $0.30_{\pm0.15}$ | $0.30_{\pm0.15}$ | $-0.86_{\pm0.40}$ | $0.29_{\pm0.15}$ | $0.29_{\pm0.15}$ | $0.29_{\pm0.15}$ | $0.29_{\pm0.15}$ |
| CBAS | $-1.86_{\pm0.00}$ | $-179.29_{\pm0.34}$ | $-179.29_{\pm0.34}$ | $-179.29_{\pm0.34}$ | $-179.29_{\pm0.34}$ | $-1.86_{\pm0.00}$ | $-180.23_{\pm0.34}$ | $-180.23_{\pm0.34}$ | $-180.23_{\pm0.34}$ | $-180.23_{\pm0.34}$ | $-1.86_{\pm0.00}$ | $-180.54_{\pm0.34}$ | $-180.54_{\pm0.34}$ | $-180.54_{\pm0.34}$ | $-180.54_{\pm0.34}$ |
| CCDDEA | $0.73_{\pm0.16}$ | $0.74_{\pm0.08}$ | $1.43_{\pm0.17}$ | $0.98_{\pm0.11}$ | $1.09_{\pm0.12}$ | $0.77_{\pm0.15}$ | $0.82_{\pm0.09}$ | $1.58_{\pm0.17}$ | $1.08_{\pm0.11}$ | $0.98_{\pm0.10}$ | $0.77_{\pm0.15}$ | $0.84_{\pm0.10}$ | $1.63_{\pm0.19}$ | $1.11_{\pm0.13}$ | $0.85_{\pm0.10}$ |
| CMAES | $0.36_{\pm0.07}$ | $0.80_{\pm0.04}$ | $0.82_{\pm0.05}$ | $0.81_{\pm0.05}$ | $0.81_{\pm0.05}$ | - | - | - | - | - | - | - | - | - | - |
| TTDDEA | $0.18_{\pm0.05}$ | $0.27_{\pm0.14}$ | $0.27_{\pm0.14}$ | $0.27_{\pm0.14}$ | $0.27_{\pm0.14}$ | $0.21_{\pm0.10}$ | $0.31_{\pm0.16}$ | $0.31_{\pm0.16}$ | $0.31_{\pm0.16}$ | $0.31_{\pm0.16}$ | $0.21_{\pm0.10}$ | $0.33_{\pm0.17}$ | $0.33_{\pm0.17}$ | $0.33_{\pm0.17}$ | $0.33_{\pm0.17}$ |
| Trimentoring | $-0.17_{\pm0.12}$ | $0.55_{\pm0.12}$ | $0.56_{\pm0.13}$ | $0.56_{\pm0.13}$ | $0.56_{\pm0.13}$ | $-0.13_{\pm0.11}$ | $0.65_{\pm0.17}$ | $0.66_{\pm0.18}$ | $0.66_{\pm0.17}$ | $0.65_{\pm0.17}$ | $-0.13_{\pm0.15}$ | $0.67_{\pm0.18}$ | $0.70_{\pm0.21}$ | $0.69_{\pm0.19}$ | $0.67_{\pm0.18}$ |

Table 57: Overall results for mujoco unconstrained tasks with 128 solutions and 0th percentile evaluations. In this case, 0% of the values are missing near the worst value and another 30% near the optimal value. Details are the same as Table 5.

| Steps | | t = 50 | | | | | t = 100 | | | | | t = 150 | | | |
|---|---|---|---|---|---|---|---|---|---|---|---|---|---|---|---|
| Task $f(x^*_{OFF})$ | | | | | | | Mujoco 1 $0.45_{\pm0.00}$ | | | | | | | | |
| Metric | FS | SI | OI | SO | SO$_\omega$ | FS | SI | OI | SO | SO$_\omega$ | FS | SI | OI | SO | SO$_\omega$ |
| ARCOO | $0.44_{\pm0.00}$ | $0.97_{\pm0.01}$ | $0.97_{\pm0.01}$ | $0.97_{\pm0.01}$ | $0.97_{\pm0.01}$ | $0.21_{\pm0.10}$ | $0.74_{\pm0.16}$ | $0.74_{\pm0.16}$ | $0.74_{\pm0.16}$ | $0.74_{\pm0.16}$ | $0.15_{\pm0.04}$ | $0.51_{\pm0.25}$ | $0.51_{\pm0.25}$ | $0.51_{\pm0.25}$ | $0.51_{\pm0.25}$ |
| BO | $0.14_{\pm0.05}$ | $0.44_{\pm0.01}$ | $0.44_{\pm0.01}$ | $0.44_{\pm0.01}$ | $0.44_{\pm0.01}$ | $0.15_{\pm0.03}$ | $0.44_{\pm0.02}$ | $0.44_{\pm0.02}$ | $0.44_{\pm0.02}$ | $0.44_{\pm0.02}$ | $0.15_{\pm0.03}$ | $0.44_{\pm0.01}$ | $0.44_{\pm0.01}$ | $0.44_{\pm0.01}$ | $0.44_{\pm0.01}$ |
| CBAS | $0.00_{\pm0.00}$ | $-43.12_{\pm0.18}$ | $-43.12_{\pm0.18}$ | $-43.12_{\pm0.18}$ | $-43.12_{\pm0.18}$ | $0.00_{\pm0.00}$ | $-43.35_{\pm0.18}$ | $-43.35_{\pm0.18}$ | $-43.35_{\pm0.18}$ | $-43.35_{\pm0.18}$ | $0.00_{\pm0.00}$ | $-43.42_{\pm0.18}$ | $-43.42_{\pm0.18}$ | $-43.42_{\pm0.18}$ | $-43.42_{\pm0.18}$ |
| CCDDEA | $0.71_{\pm0.09}$ | $0.69_{\pm0.07}$ | $1.30_{\pm0.17}$ | $0.90_{\pm0.09}$ | $1.00_{\pm0.10}$ | $0.73_{\pm0.08}$ | $0.78_{\pm0.05}$ | $1.47_{\pm0.15}$ | $1.02_{\pm0.06}$ | $0.93_{\pm0.05}$ | $0.73_{\pm0.08}$ | $0.81_{\pm0.06}$ | $1.52_{\pm0.16}$ | $1.06_{\pm0.07}$ | $0.81_{\pm0.06}$ |
| CMAES | $0.39_{\pm0.08}$ | $0.74_{\pm0.03}$ | $0.76_{\pm0.06}$ | $0.75_{\pm0.04}$ | $0.76_{\pm0.05}$ | - | - | - | - | - | - | - | - | - | - |
| TTDDEA | $0.20_{\pm0.00}$ | $-0.12_{\pm0.23}$ | $-0.12_{\pm0.23}$ | $-0.12_{\pm0.23}$ | $-0.12_{\pm0.23}$ | $0.23_{\pm0.07}$ | $0.01_{\pm0.26}$ | $0.01_{\pm0.26}$ | $0.01_{\pm0.26}$ | $0.01_{\pm0.26}$ | $0.21_{\pm0.10}$ | $0.05_{\pm0.27}$ | $0.05_{\pm0.27}$ | $0.05_{\pm0.27}$ | $0.05_{\pm0.27}$ |
| Trimentoring | $0.30_{\pm0.04}$ | $0.72_{\pm0.04}$ | $0.72_{\pm0.05}$ | $0.72_{\pm0.05}$ | $0.72_{\pm0.05}$ | $0.35_{\pm0.06}$ | $0.75_{\pm0.05}$ | $0.77_{\pm0.07}$ | $0.76_{\pm0.05}$ | $0.75_{\pm0.05}$ | $0.37_{\pm0.07}$ | $0.77_{\pm0.05}$ | $0.79_{\pm0.09}$ | $0.78_{\pm0.07}$ | $0.77_{\pm0.05}$ |
| Task $f(x^*_{OFF})$ | | | | | | | Mujoco 2 $0.03_{\pm0.01}$ | | | | | | | | |
| ARCOO | $0.02_{\pm0.00}$ | $0.99_{\pm0.01}$ | $0.99_{\pm0.01}$ | $0.99_{\pm0.01}$ | $0.99_{\pm0.01}$ | $-0.60_{\pm0.23}$ | $0.71_{\pm0.18}$ | $0.71_{\pm0.18}$ | $0.71_{\pm0.18}$ | $0.71_{\pm0.18}$ | $-0.79_{\pm0.30}$ | $0.38_{\pm0.25}$ | $0.38_{\pm0.25}$ | $0.38_{\pm0.25}$ | $0.38_{\pm0.25}$ |
| BO | $-0.85_{\pm0.16}$ | $0.36_{\pm0.16}$ | $0.36_{\pm0.16}$ | $0.36_{\pm0.16}$ | $0.36_{\pm0.16}$ | $-0.93_{\pm0.24}$ | $0.36_{\pm0.17}$ | $0.36_{\pm0.17}$ | $0.36_{\pm0.17}$ | $0.36_{\pm0.17}$ | $-0.86_{\pm0.43}$ | $0.35_{\pm0.17}$ | $0.35_{\pm0.17}$ | $0.35_{\pm0.17}$ | $0.35_{\pm0.17}$ |
| CBAS | $-1.86_{\pm0.00}$ | $-185.68_{\pm0.45}$ | $-185.68_{\pm0.45}$ | $-185.68_{\pm0.45}$ | $-185.68_{\pm0.45}$ | $-1.86_{\pm0.00}$ | $-186.65_{\pm0.46}$ | $-186.65_{\pm0.46}$ | $-186.65_{\pm0.46}$ | $-186.65_{\pm0.46}$ | $-1.86_{\pm0.00}$ | $-186.97_{\pm0.46}$ | $-186.97_{\pm0.46}$ | $-186.97_{\pm0.46}$ | $-186.97_{\pm0.46}$ |
| CCDDEA | $0.12_{\pm0.29}$ | $0.72_{\pm0.06}$ | $1.33_{\pm0.07}$ | $0.93_{\pm0.05}$ | $1.04_{\pm0.04}$ | $0.34_{\pm0.14}$ | $0.74_{\pm0.06}$ | $1.40_{\pm0.09}$ | $0.96_{\pm0.05}$ | $0.88_{\pm0.05}$ | $0.34_{\pm0.14}$ | $0.75_{\pm0.08}$ | $1.43_{\pm0.13}$ | $0.98_{\pm0.09}$ | $0.76_{\pm0.08}$ |
| CMAES | $-0.23_{\pm0.11}$ | $0.59_{\pm0.07}$ | $0.59_{\pm0.07}$ | $0.59_{\pm0.07}$ | $0.59_{\pm0.07}$ | - | - | - | - | - | - | - | - | - | - |
| TTDDEA | $-1.86_{\pm0.00}$ | $-0.08_{\pm0.34}$ | $-0.08_{\pm0.34}$ | $-0.08_{\pm0.34}$ | $-0.08_{\pm0.34}$ | $-1.86_{\pm0.00}$ | $-0.10_{\pm0.36}$ | $-0.10_{\pm0.36}$ | $-0.10_{\pm0.36}$ | $-0.10_{\pm0.36}$ | $-1.86_{\pm0.00}$ | $-0.09_{\pm0.35}$ | $-0.09_{\pm0.35}$ | $-0.09_{\pm0.35}$ | $-0.09_{\pm0.35}$ |
| Trimentoring | $-0.13_{\pm0.12}$ | $0.54_{\pm0.11}$ | $0.54_{\pm0.11}$ | $0.54_{\pm0.11}$ | $0.54_{\pm0.11}$ | $-0.14_{\pm0.11}$ | $0.67_{\pm0.08}$ | $0.68_{\pm0.08}$ | $0.67_{\pm0.08}$ | $0.67_{\pm0.08}$ | $-0.12_{\pm0.06}$ | $0.70_{\pm0.08}$ | $0.72_{\pm0.08}$ | $0.71_{\pm0.08}$ | $0.70_{\pm0.08}$ |

Table 58: Overall results for mujoco unconstrained tasks with 128 solutions and 0th percentile evaluations. In this case, 0% of the values are missing near the worst value and another 20% near the optimal value. Details are the same as Table 5.

| Steps | | t = 50 | | | | | t = 100 | | | | | t = 150 | | | |
|---|---|---|---|---|---|---|---|---|---|---|---|---|---|---|---|
| Task $f(x^*_{OFF})$ | | | | | | | Mujoco 1 $0.5_{\pm0.00}$ | | | | | | | | |
| Metric | FS | SI | OI | SO | SO$_\omega$ | FS | SI | OI | SO | SO$_\omega$ | FS | SI | OI | SO | SO$_\omega$ |
| ARCOO | $0.49_{\pm0.00}$ | $0.96_{\pm0.01}$ | $0.96_{\pm0.01}$ | $0.96_{\pm0.01}$ | $0.96_{\pm0.01}$ | $0.21_{\pm0.06}$ | $0.62_{\pm0.17}$ | $0.62_{\pm0.17}$ | $0.62_{\pm0.17}$ | $0.62_{\pm0.17}$ | $0.16_{\pm0.07}$ | $0.28_{\pm0.23}$ | $0.28_{\pm0.23}$ | $0.28_{\pm0.23}$ | $0.28_{\pm0.23}$ |
| BO | $0.13_{\pm0.05}$ | $0.27_{\pm0.02}$ | $0.27_{\pm0.02}$ | $0.27_{\pm0.02}$ | $0.27_{\pm0.02}$ | $0.15_{\pm0.03}$ | $0.26_{\pm0.01}$ | $0.26_{\pm0.01}$ | $0.26_{\pm0.01}$ | $0.26_{\pm0.01}$ | $0.16_{\pm0.04}$ | $0.26_{\pm0.01}$ | $0.26_{\pm0.01}$ | $0.26_{\pm0.01}$ | $0.26_{\pm0.01}$ |
| CBAS | $0.00_{\pm0.00}$ | $-48.64_{\pm0.34}$ | $-48.64_{\pm0.34}$ | $-48.64_{\pm0.34}$ | $-48.64_{\pm0.34}$ | $0.00_{\pm0.00}$ | $-48.89_{\pm0.34}$ | $-48.89_{\pm0.34}$ | $-48.89_{\pm0.34}$ | $-48.89_{\pm0.34}$ | $0.00_{\pm0.00}$ | $-48.98_{\pm0.34}$ | $-48.98_{\pm0.34}$ | $-48.98_{\pm0.34}$ | $-48.98_{\pm0.34}$ |
| CCDDEA | $0.75_{\pm0.11}$ | $0.72_{\pm0.05}$ | $1.32_{\pm0.17}$ | $0.93_{\pm0.08}$ | $1.03_{\pm0.09}$ | $0.75_{\pm0.11}$ | $0.79_{\pm0.06}$ | $1.43_{\pm0.16}$ | $1.01_{\pm0.08}$ | $0.93_{\pm0.07}$ | $0.75_{\pm0.11}$ | $0.81_{\pm0.08}$ | $1.47_{\pm0.18}$ | $1.04_{\pm0.10}$ | $0.81_{\pm0.08}$ |
| CMAES | $0.42_{\pm0.08}$ | $0.61_{\pm0.07}$ | $0.62_{\pm0.07}$ | $0.61_{\pm0.07}$ | $0.62_{\pm0.07}$ | - | - | - | - | - | - | - | - | - | - |
| TTDDEA | $0.19_{\pm0.00}$ | $-0.77_{\pm0.76}$ | $-0.77_{\pm0.76}$ | $-0.77_{\pm0.76}$ | $-0.77_{\pm0.76}$ | $0.27_{\pm0.11}$ | $-0.58_{\pm0.73}$ | $-0.58_{\pm0.73}$ | $-0.58_{\pm0.73}$ | $-0.58_{\pm0.73}$ | $0.27_{\pm0.00}$ | $-0.52_{\pm0.74}$ | $-0.52_{\pm0.74}$ | $-0.52_{\pm0.74}$ | $-0.52_{\pm0.74}$ |
| Trimentoring | $0.32_{\pm0.07}$ | $0.57_{\pm0.10}$ | $0.57_{\pm0.10}$ | $0.57_{\pm0.10}$ | $0.57_{\pm0.10}$ | $0.36_{\pm0.04}$ | $0.62_{\pm0.08}$ | $0.62_{\pm0.08}$ | $0.62_{\pm0.08}$ | $0.62_{\pm0.08}$ | $0.37_{\pm0.04}$ | $0.65_{\pm0.08}$ | $0.65_{\pm0.08}$ | $0.65_{\pm0.08}$ | $0.65_{\pm0.08}$ |
| Task $f(x^*_{OFF})$ | | | | | | | Mujoco 2 $0.11_{\pm0.00}$ | | | | | | | | |
| ARCOO | $0.10_{\pm0.00}$ | $0.97_{\pm0.01}$ | $0.97_{\pm0.01}$ | $0.97_{\pm0.01}$ | $0.97_{\pm0.01}$ | $-0.53_{\pm0.34}$ | $0.68_{\pm0.12}$ | $0.68_{\pm0.12}$ | $0.68_{\pm0.12}$ | $0.68_{\pm0.12}$ | $-0.67_{\pm0.27}$ | $0.21_{\pm0.26}$ | $0.21_{\pm0.26}$ | $0.21_{\pm0.26}$ | $0.21_{\pm0.26}$ |
| BO | $-0.62_{\pm0.20}$ | $0.07_{\pm0.21}$ | $0.07_{\pm0.21}$ | $0.07_{\pm0.21}$ | $0.07_{\pm0.21}$ | $-0.99_{\pm0.52}$ | $0.05_{\pm0.22}$ | $0.05_{\pm0.22}$ | $0.05_{\pm0.22}$ | $0.05_{\pm0.22}$ | $-0.78_{\pm0.42}$ | $0.04_{\pm0.21}$ | $0.04_{\pm0.21}$ | $0.04_{\pm0.21}$ | $0.04_{\pm0.21}$ |
| CBAS | $-1.86_{\pm0.00}$ | $-194.21_{\pm0.35}$ | $-194.21_{\pm0.35}$ | $-194.21_{\pm0.35}$ | $-194.21_{\pm0.35}$ | $-1.86_{\pm0.00}$ | $-195.23_{\pm0.35}$ | $-195.23_{\pm0.35}$ | $-195.23_{\pm0.35}$ | $-195.23_{\pm0.35}$ | $-1.86_{\pm0.00}$ | $-195.56_{\pm0.35}$ | $-195.56_{\pm0.35}$ | $-195.56_{\pm0.35}$ | $-195.56_{\pm0.35}$ |
| CCDDEA | $0.12_{\pm0.17}$ | $0.70_{\pm0.04}$ | $1.24_{\pm0.09}$ | $0.90_{\pm0.04}$ | $0.99_{\pm0.05}$ | $0.38_{\pm0.09}$ | $0.69_{\pm0.06}$ | $1.35_{\pm0.09}$ | $0.92_{\pm0.07}$ | $0.83_{\pm0.06}$ | $0.38_{\pm0.09}$ | $0.73_{\pm0.08}$ | $1.41_{\pm0.10}$ | $0.96_{\pm0.09}$ | $0.73_{\pm0.08}$ |
| CMAES | $-0.26_{\pm0.11}$ | $0.40_{\pm0.11}$ | $0.40_{\pm0.11}$ | $0.40_{\pm0.11}$ | $0.40_{\pm0.11}$ | - | - | - | - | - | - | - | - | - | - |
| TTDDEA | $-1.86_{\pm0.00}$ | $-1.13_{\pm0.61}$ | $-1.13_{\pm0.61}$ | $-1.13_{\pm0.61}$ | $-1.13_{\pm0.61}$ | $-1.86_{\pm0.00}$ | $-1.14_{\pm0.61}$ | $-1.14_{\pm0.61}$ | $-1.14_{\pm0.61}$ | $-1.14_{\pm0.61}$ | $-1.86_{\pm0.00}$ | $-1.15_{\pm0.62}$ | $-1.15_{\pm0.62}$ | $-1.15_{\pm0.62}$ | $-1.15_{\pm0.62}$ |
| Trimentoring | $-0.15_{\pm0.10}$ | $0.42_{\pm0.11}$ | $0.42_{\pm0.11}$ | $0.42_{\pm0.11}$ | $0.42_{\pm0.11}$ | $-0.13_{\pm0.07}$ | $0.50_{\pm0.09}$ | $0.50_{\pm0.09}$ | $0.50_{\pm0.09}$ | $0.50_{\pm0.09}$ | $-0.15_{\pm0.11}$ | $0.53_{\pm0.09}$ | $0.53_{\pm0.09}$ | $0.53_{\pm0.09}$ | $0.53_{\pm0.09}$ |

Table 59: Overall results for mujoco unconstrained tasks with 128 solutions and 0th percentile evaluations. In this case, 0% of the values are missing near the worst value and another 10% near the optimal value. Details are the same as Table 5.

| Steps | t = 50 | | | | | t = 100 | | | | | t = 150 | | | | |
|---|---|---|---|---|---|---|---|---|---|---|---|---|---|---|---|
| **Task** $f(x^*_{OFF})$ | Mujoco 1  $0.59_{\pm0.01}$ | | | | | | | | | | | | | | |
| Metric | FS | SI | OI | SO | $SO_\omega$ | FS | SI | OI | SO | $SO_\omega$ | FS | SI | OI | SO | $SO_\omega$ |
| ARCOO | $0.56_{\pm0.05}$ | $0.92_{\pm0.02}$ | $0.92_{\pm0.02}$ | $0.92_{\pm0.02}$ | $0.92_{\pm0.02}$ | $0.20_{\pm0.06}$ | $0.63_{\pm0.08}$ | $0.63_{\pm0.08}$ | $0.63_{\pm0.08}$ | $0.63_{\pm0.08}$ | $0.18_{\pm0.02}$ | $0.27_{\pm0.10}$ | $0.27_{\pm0.10}$ | $0.27_{\pm0.10}$ | $0.27_{\pm0.10}$ |
| BO | $0.15_{\pm0.04}$ | $-0.12_{\pm0.04}$ | $-0.12_{\pm0.04}$ | $-0.12_{\pm0.04}$ | $-0.12_{\pm0.04}$ | $0.14_{\pm0.05}$ | $-0.14_{\pm0.04}$ | $-0.14_{\pm0.04}$ | $-0.14_{\pm0.04}$ | $-0.14_{\pm0.04}$ | $0.16_{\pm0.04}$ | $-0.14_{\pm0.04}$ | $-0.14_{\pm0.04}$ | $-0.14_{\pm0.04}$ | $-0.14_{\pm0.04}$ |
| CBAS | $0.00_{\pm0.00}$ | $-57.79_{\pm0.58}$ | $-57.79_{\pm0.58}$ | $-57.79_{\pm0.58}$ | $-57.79_{\pm0.58}$ | $0.00_{\pm0.00}$ | $-58.10_{\pm0.58}$ | $-58.10_{\pm0.58}$ | $-58.10_{\pm0.58}$ | $-58.10_{\pm0.58}$ | $0.00_{\pm0.00}$ | $-58.20_{\pm0.59}$ | $-58.20_{\pm0.59}$ | $-58.20_{\pm0.59}$ | $-58.20_{\pm0.59}$ |
| CCDDEA | $0.71_{\pm0.07}$ | $0.62_{\pm0.10}$ | $1.15_{\pm0.22}$ | $0.80_{\pm0.13}$ | $0.89_{\pm0.15}$ | $0.70_{\pm0.06}$ | $0.67_{\pm0.08}$ | $1.24_{\pm0.10}$ | $0.87_{\pm0.08}$ | $0.80_{\pm0.08}$ | $0.70_{\pm0.06}$ | $0.69_{\pm0.10}$ | $1.27_{\pm0.11}$ | $0.90_{\pm0.10}$ | $0.69_{\pm0.10}$ |
| CMAES | $0.43_{\pm0.11}$ | $0.33_{\pm0.13}$ | $0.33_{\pm0.13}$ | $0.33_{\pm0.13}$ | $0.33_{\pm0.13}$ | - | | | | | - | | | | |
| TTDDEA | $0.34_{\pm0.12}$ | $-6.64_{\pm4.59}$ | $-6.64_{\pm4.59}$ | $-6.64_{\pm4.59}$ | $-6.64_{\pm4.59}$ | $0.34_{\pm0.12}$ | $-5.95_{\pm4.73}$ | $-5.95_{\pm4.73}$ | $-5.95_{\pm4.73}$ | $-5.95_{\pm4.73}$ | $0.34_{\pm0.12}$ | $-5.62_{\pm4.80}$ | $-5.62_{\pm4.80}$ | $-5.62_{\pm4.80}$ | $-5.62_{\pm4.80}$ |
| Trimentoring | $0.30_{\pm0.05}$ | $0.24_{\pm0.13}$ | $0.24_{\pm0.13}$ | $0.24_{\pm0.13}$ | $0.24_{\pm0.13}$ | $0.34_{\pm0.08}$ | $0.26_{\pm0.13}$ | $0.26_{\pm0.13}$ | $0.26_{\pm0.13}$ | $0.26_{\pm0.13}$ | $0.32_{\pm0.09}$ | $0.28_{\pm0.15}$ | $0.28_{\pm0.15}$ | $0.28_{\pm0.15}$ | $0.28_{\pm0.15}$ |
| **Task** $f(x^*_{OFF})$ | Mujoco 2  $0.23_{\pm0.01}$ | | | | | | | | | | | | | | |
| ARCOO | $0.21_{\pm0.01}$ | $0.93_{\pm0.04}$ | $0.93_{\pm0.04}$ | $0.93_{\pm0.04}$ | $0.93_{\pm0.04}$ | $-0.32_{\pm0.09}$ | $0.60_{\pm0.15}$ | $0.60_{\pm0.15}$ | $0.60_{\pm0.15}$ | $0.60_{\pm0.15}$ | $-0.76_{\pm0.27}$ | $-0.20_{\pm0.48}$ | $-0.20_{\pm0.48}$ | $-0.20_{\pm0.48}$ | $-0.20_{\pm0.48}$ |
| BO | $-0.62_{\pm0.23}$ | $-0.06_{\pm0.27}$ | $-0.06_{\pm0.27}$ | $-0.06_{\pm0.27}$ | $-0.06_{\pm0.27}$ | $-0.90_{\pm0.56}$ | $-0.10_{\pm0.28}$ | $-0.10_{\pm0.28}$ | $-0.10_{\pm0.28}$ | $-0.10_{\pm0.28}$ | $-0.76_{\pm0.24}$ | $-0.10_{\pm0.28}$ | $-0.10_{\pm0.28}$ | $-0.10_{\pm0.28}$ | $-0.10_{\pm0.28}$ |
| CBAS | $-1.86_{\pm0.00}$ | $-205.86_{\pm0.46}$ | $-205.86_{\pm0.46}$ | $-205.86_{\pm0.46}$ | $-205.86_{\pm0.46}$ | $-1.86_{\pm0.00}$ | $-206.94_{\pm0.46}$ | $-206.94_{\pm0.46}$ | $-206.94_{\pm0.46}$ | $-206.94_{\pm0.46}$ | $-1.86_{\pm0.00}$ | $-207.29_{\pm0.46}$ | $-207.29_{\pm0.46}$ | $-207.29_{\pm0.46}$ | $-207.29_{\pm0.46}$ |
| CCDDEA | $0.00_{\pm0.23}$ | $0.60_{\pm0.11}$ | $1.11_{\pm0.23}$ | $0.78_{\pm0.14}$ | $0.86_{\pm0.16}$ | $0.38_{\pm0.11}$ | $0.65_{\pm0.09}$ | $1.20_{\pm0.14}$ | $0.84_{\pm0.10}$ | $0.77_{\pm0.10}$ | $0.38_{\pm0.11}$ | $0.69_{\pm0.11}$ | $1.26_{\pm0.16}$ | $0.89_{\pm0.13}$ | $0.69_{\pm0.11}$ |
| CMAES | $-0.27_{\pm0.10}$ | $0.14_{\pm0.20}$ | $0.14_{\pm0.20}$ | $0.14_{\pm0.20}$ | $0.14_{\pm0.20}$ | - | | | | | - | | | | |
| TTDDEA | $-1.46_{\pm0.70}$ | $-2.12_{\pm1.74}$ | $-2.12_{\pm1.74}$ | $-2.12_{\pm1.74}$ | $-2.12_{\pm1.74}$ | $-1.10_{\pm0.88}$ | $-1.93_{\pm1.90}$ | $-1.93_{\pm1.90}$ | $-1.93_{\pm1.90}$ | $-1.93_{\pm1.90}$ | $-1.24_{\pm0.84}$ | $-1.88_{\pm1.94}$ | $-1.88_{\pm1.94}$ | $-1.88_{\pm1.94}$ | $-1.88_{\pm1.94}$ |
| Trimentoring | $-0.19_{\pm0.04}$ | $0.10_{\pm0.12}$ | $0.10_{\pm0.12}$ | $0.10_{\pm0.12}$ | $0.10_{\pm0.12}$ | $-0.12_{\pm0.07}$ | $0.18_{\pm0.10}$ | $0.18_{\pm0.10}$ | $0.18_{\pm0.10}$ | $0.18_{\pm0.10}$ | $-0.13_{\pm0.09}$ | $0.21_{\pm0.12}$ | $0.21_{\pm0.12}$ | $0.21_{\pm0.12}$ | $0.21_{\pm0.12}$ |

Table 60: Overall results for mujoco unconstrained tasks with 128 solutions and 0th percentile evaluations. In this case, 10% of the values are missing near the worst value and another 40% near the optimal value. Details are the same as Table 5.

| Steps | t = 50 | | | | | t = 100 | | | | | t = 150 | | | | |
|---|---|---|---|---|---|---|---|---|---|---|---|---|---|---|---|
| **Task** $f(x^*_{OFF})$ | Mujoco 1  $0.4_{\pm0.00}$ | | | | | | | | | | | | | | |
| Metric | FS | SI | OI | SO | $SO_\omega$ | FS | SI | OI | SO | $SO_\omega$ | FS | SI | OI | SO | $SO_\omega$ |
| ARCOO | $0.32_{\pm0.11}$ | $0.98_{\pm0.01}$ | $0.98_{\pm0.01}$ | $0.98_{\pm0.01}$ | $0.98_{\pm0.01}$ | $0.16_{\pm0.05}$ | $0.78_{\pm0.06}$ | $0.78_{\pm0.06}$ | $0.78_{\pm0.06}$ | $0.78_{\pm0.06}$ | $0.13_{\pm0.04}$ | $0.65_{\pm0.03}$ | $0.65_{\pm0.03}$ | $0.65_{\pm0.03}$ | $0.65_{\pm0.03}$ |
| BO | $0.13_{\pm0.03}$ | $0.56_{\pm0.01}$ | $0.56_{\pm0.01}$ | $0.56_{\pm0.01}$ | $0.56_{\pm0.01}$ | $0.15_{\pm0.03}$ | $0.56_{\pm0.01}$ | $0.56_{\pm0.01}$ | $0.56_{\pm0.01}$ | $0.56_{\pm0.01}$ | $0.15_{\pm0.06}$ | $0.56_{\pm0.01}$ | $0.56_{\pm0.01}$ | $0.56_{\pm0.01}$ | $0.56_{\pm0.01}$ |
| CBAS | $0.00_{\pm0.00}$ | $-38.87_{\pm0.22}$ | $-38.87_{\pm0.22}$ | $-38.87_{\pm0.22}$ | $-38.87_{\pm0.22}$ | $0.00_{\pm0.00}$ | $-39.08_{\pm0.22}$ | $-39.08_{\pm0.22}$ | $-39.08_{\pm0.22}$ | $-39.08_{\pm0.22}$ | $0.00_{\pm0.00}$ | $-39.15_{\pm0.22}$ | $-39.15_{\pm0.22}$ | $-39.15_{\pm0.22}$ | $-39.15_{\pm0.22}$ |
| CCDDEA | $0.71_{\pm0.20}$ | $0.72_{\pm0.05}$ | $1.34_{\pm0.14}$ | $0.93_{\pm0.07}$ | $1.04_{\pm0.08}$ | $0.74_{\pm0.10}$ | $0.80_{\pm0.04}$ | $1.49_{\pm0.14}$ | $1.04_{\pm0.06}$ | $0.95_{\pm0.05}$ | $0.74_{\pm0.10}$ | $0.83_{\pm0.05}$ | $1.54_{\pm0.15}$ | $1.07_{\pm0.07}$ | $0.83_{\pm0.05}$ |
| CMAES | $0.29_{\pm0.06}$ | $0.76_{\pm0.06}$ | $0.77_{\pm0.07}$ | $0.77_{\pm0.06}$ | $0.77_{\pm0.07}$ | - | | | | | - | | | | |
| TTDDEA | $0.20_{\pm0.11}$ | $0.30_{\pm0.36}$ | $0.30_{\pm0.36}$ | $0.30_{\pm0.36}$ | $0.30_{\pm0.36}$ | $0.21_{\pm0.10}$ | $0.36_{\pm0.37}$ | $0.36_{\pm0.37}$ | $0.36_{\pm0.37}$ | $0.36_{\pm0.37}$ | $0.17_{\pm0.08}$ | $0.38_{\pm0.37}$ | $0.38_{\pm0.37}$ | $0.38_{\pm0.37}$ | $0.38_{\pm0.37}$ |
| Trimentoring | $0.30_{\pm0.06}$ | $0.72_{\pm0.07}$ | $0.74_{\pm0.09}$ | $0.73_{\pm0.07}$ | $0.73_{\pm0.08}$ | $0.32_{\pm0.11}$ | $0.77_{\pm0.08}$ | $0.80_{\pm0.12}$ | $0.78_{\pm0.10}$ | $0.78_{\pm0.09}$ | $0.34_{\pm0.09}$ | $0.80_{\pm0.08}$ | $0.83_{\pm0.13}$ | $0.81_{\pm0.10}$ | $0.80_{\pm0.08}$ |
| **Task** $f(x^*_{OFF})$ | Mujoco 2  $-0.04_{\pm0.00}$ | | | | | | | | | | | | | | |
| ARCOO | $-0.09_{\pm0.11}$ | $0.98_{\pm0.01}$ | $0.98_{\pm0.01}$ | $0.98_{\pm0.01}$ | $0.98_{\pm0.01}$ | $-0.56_{\pm0.12}$ | $0.71_{\pm0.17}$ | $0.71_{\pm0.17}$ | $0.71_{\pm0.17}$ | $0.71_{\pm0.17}$ | $-0.71_{\pm0.46}$ | $0.45_{\pm0.27}$ | $0.45_{\pm0.27}$ | $0.45_{\pm0.27}$ | $0.45_{\pm0.27}$ |
| BO | $-0.92_{\pm0.56}$ | $0.25_{\pm0.15}$ | $0.25_{\pm0.15}$ | $0.25_{\pm0.15}$ | $0.25_{\pm0.15}$ | $-0.78_{\pm0.33}$ | $0.26_{\pm0.14}$ | $0.26_{\pm0.14}$ | $0.26_{\pm0.14}$ | $0.26_{\pm0.14}$ | $-0.94_{\pm0.54}$ | $0.27_{\pm0.13}$ | $0.27_{\pm0.13}$ | $0.27_{\pm0.13}$ | $0.27_{\pm0.13}$ |
| CBAS | $-1.86_{\pm0.00}$ | $-179.29_{\pm0.34}$ | $-179.29_{\pm0.34}$ | $-179.29_{\pm0.34}$ | $-179.29_{\pm0.34}$ | $-1.86_{\pm0.00}$ | $-180.23_{\pm0.34}$ | $-180.23_{\pm0.34}$ | $-180.23_{\pm0.34}$ | $-180.23_{\pm0.34}$ | $-1.86_{\pm0.00}$ | $-180.54_{\pm0.34}$ | $-180.54_{\pm0.34}$ | $-180.54_{\pm0.34}$ | $-180.54_{\pm0.34}$ |
| CCDDEA | $0.26_{\pm0.17}$ | $0.75_{\pm0.05}$ | $1.43_{\pm0.11}$ | $0.98_{\pm0.06}$ | $1.10_{\pm0.07}$ | $0.26_{\pm0.13}$ | $0.76_{\pm0.06}$ | $1.45_{\pm0.12}$ | $0.99_{\pm0.08}$ | $0.90_{\pm0.07}$ | $0.26_{\pm0.13}$ | $0.76_{\pm0.08}$ | $1.45_{\pm0.15}$ | $0.99_{\pm0.10}$ | $0.76_{\pm0.08}$ |
| CMAES | $-0.26_{\pm0.03}$ | $0.64_{\pm0.06}$ | $0.64_{\pm0.06}$ | $0.64_{\pm0.06}$ | $0.64_{\pm0.06}$ | - | | | | | - | | | | |
| TTDDEA | $-0.79_{\pm0.44}$ | $-0.27_{\pm0.27}$ | $-0.31_{\pm0.34}$ | $-0.28_{\pm0.29}$ | $-0.29_{\pm0.30}$ | $-0.84_{\pm0.43}$ | $-0.09_{\pm0.22}$ | $-0.10_{\pm0.23}$ | $-0.09_{\pm0.22}$ | $-0.09_{\pm0.22}$ | $-0.47_{\pm0.38}$ | $-0.01_{\pm0.23}$ | $-0.02_{\pm0.24}$ | $-0.01_{\pm0.23}$ | $-0.01_{\pm0.23}$ |
| Trimentoring | $-0.22_{\pm0.07}$ | $0.58_{\pm0.15}$ | $0.58_{\pm0.15}$ | $0.58_{\pm0.15}$ | $0.58_{\pm0.15}$ | $-0.16_{\pm0.07}$ | $0.66_{\pm0.14}$ | $0.66_{\pm0.14}$ | $0.66_{\pm0.14}$ | $0.66_{\pm0.14}$ | $-0.17_{\pm0.12}$ | $0.71_{\pm0.13}$ | $0.71_{\pm0.13}$ | $0.71_{\pm0.13}$ | $0.71_{\pm0.13}$ |

Table 61: Overall results for mujoco unconstrained tasks with 128 solutions and 0th percentile evaluations. In this case, 20% of the values are missing near the worst value and another 30% near the optimal value. Details are the same as Table 5.

| Steps | t = 50 | | | | | t = 100 | | | | | t = 150 | | | | |
|---|---|---|---|---|---|---|---|---|---|---|---|---|---|---|---|
| **Task** $f(x^*_{OFF})$ | Mujoco 1  $0.45_{\pm0.00}$ | | | | | | | | | | | | | | |
| Metric | FS | SI | OI | SO | $SO_\omega$ | FS | SI | OI | SO | $SO_\omega$ | FS | SI | OI | SO | $SO_\omega$ |
| ARCOO | $0.38_{\pm0.06}$ | $0.97_{\pm0.02}$ | $0.97_{\pm0.02}$ | $0.97_{\pm0.02}$ | $0.97_{\pm0.02}$ | $0.16_{\pm0.06}$ | $0.66_{\pm0.09}$ | $0.66_{\pm0.09}$ | $0.66_{\pm0.09}$ | $0.66_{\pm0.09}$ | $0.14_{\pm0.03}$ | $0.52_{\pm0.10}$ | $0.52_{\pm0.10}$ | $0.52_{\pm0.10}$ | $0.52_{\pm0.10}$ |
| BO | $0.14_{\pm0.03}$ | $0.44_{\pm0.02}$ | $0.44_{\pm0.02}$ | $0.44_{\pm0.02}$ | $0.44_{\pm0.02}$ | $0.16_{\pm0.03}$ | $0.44_{\pm0.01}$ | $0.44_{\pm0.01}$ | $0.44_{\pm0.01}$ | $0.44_{\pm0.01}$ | $0.15_{\pm0.02}$ | $0.44_{\pm0.01}$ | $0.44_{\pm0.01}$ | $0.44_{\pm0.01}$ | $0.44_{\pm0.01}$ |
| CBAS | $0.00_{\pm0.00}$ | $-43.12_{\pm0.18}$ | $-43.12_{\pm0.18}$ | $-43.12_{\pm0.18}$ | $-43.12_{\pm0.18}$ | $0.00_{\pm0.00}$ | $-43.35_{\pm0.18}$ | $-43.35_{\pm0.18}$ | $-43.35_{\pm0.18}$ | $-43.35_{\pm0.18}$ | $0.00_{\pm0.00}$ | $-43.42_{\pm0.18}$ | $-43.42_{\pm0.18}$ | $-43.42_{\pm0.18}$ | $-43.42_{\pm0.18}$ |
| CCDDEA | $0.65_{\pm0.14}$ | $0.73_{\pm0.06}$ | $1.33_{\pm0.12}$ | $0.94_{\pm0.07}$ | $1.04_{\pm0.08}$ | $0.65_{\pm0.14}$ | $0.76_{\pm0.10}$ | $1.39_{\pm0.17}$ | $0.98_{\pm0.12}$ | $0.90_{\pm0.12}$ | $0.65_{\pm0.14}$ | $0.77_{\pm0.13}$ | $1.41_{\pm0.21}$ | $1.00_{\pm0.15}$ | $0.77_{\pm0.13}$ |
| CMAES | $0.31_{\pm0.07}$ | $0.67_{\pm0.07}$ | $0.67_{\pm0.07}$ | $0.67_{\pm0.07}$ | $0.67_{\pm0.07}$ | - | | | | | - | | | | |
| TTDDEA | $0.23_{\pm0.08}$ | $0.19_{\pm0.27}$ | $0.19_{\pm0.27}$ | $0.19_{\pm0.27}$ | $0.19_{\pm0.27}$ | $0.28_{\pm0.06}$ | $0.29_{\pm0.23}$ | $0.29_{\pm0.23}$ | $0.29_{\pm0.23}$ | $0.29_{\pm0.23}$ | $0.24_{\pm0.04}$ | $0.34_{\pm0.23}$ | $0.34_{\pm0.23}$ | $0.34_{\pm0.23}$ | $0.34_{\pm0.23}$ |
| Trimentoring | $0.28_{\pm0.12}$ | $0.65_{\pm0.06}$ | $0.66_{\pm0.07}$ | $0.66_{\pm0.06}$ | $0.66_{\pm0.06}$ | $0.31_{\pm0.11}$ | $0.69_{\pm0.11}$ | $0.69_{\pm0.12}$ | $0.69_{\pm0.12}$ | $0.69_{\pm0.12}$ | $0.31_{\pm0.11}$ | $0.70_{\pm0.14}$ | $0.71_{\pm0.15}$ | $0.71_{\pm0.14}$ | $0.70_{\pm0.14}$ |
| **Task** $f(x^*_{OFF})$ | Mujoco 2  $0.03_{\pm0.01}$ | | | | | | | | | | | | | | |
| ARCOO | $0.02_{\pm0.00}$ | $0.99_{\pm0.01}$ | $0.99_{\pm0.01}$ | $0.99_{\pm0.01}$ | $0.99_{\pm0.01}$ | $-0.77_{\pm0.27}$ | $0.60_{\pm0.22}$ | $0.60_{\pm0.22}$ | $0.60_{\pm0.22}$ | $0.60_{\pm0.22}$ | $-0.99_{\pm0.42}$ | $0.10_{\pm0.47}$ | $0.10_{\pm0.47}$ | $0.10_{\pm0.47}$ | $0.10_{\pm0.47}$ |
| BO | $-0.99_{\pm0.41}$ | $0.11_{\pm0.21}$ | $0.11_{\pm0.21}$ | $0.11_{\pm0.21}$ | $0.11_{\pm0.21}$ | $-0.95_{\pm0.55}$ | $0.11_{\pm0.22}$ | $0.11_{\pm0.22}$ | $0.11_{\pm0.22}$ | $0.11_{\pm0.22}$ | $-0.98_{\pm0.52}$ | $0.11_{\pm0.22}$ | $0.11_{\pm0.22}$ | $0.11_{\pm0.22}$ | $0.11_{\pm0.22}$ |
| CBAS | $-1.86_{\pm0.00}$ | $-185.68_{\pm0.45}$ | $-185.68_{\pm0.45}$ | $-185.68_{\pm0.45}$ | $-185.68_{\pm0.45}$ | $-1.86_{\pm0.00}$ | $-186.65_{\pm0.46}$ | $-186.65_{\pm0.46}$ | $-186.65_{\pm0.46}$ | $-186.65_{\pm0.46}$ | $-1.86_{\pm0.00}$ | $-186.97_{\pm0.46}$ | $-186.97_{\pm0.46}$ | $-186.97_{\pm0.46}$ | $-186.97_{\pm0.46}$ |
| CCDDEA | $0.19_{\pm0.25}$ | $0.73_{\pm0.05}$ | $1.26_{\pm0.09}$ | $0.92_{\pm0.06}$ | $1.01_{\pm0.07}$ | $0.26_{\pm0.14}$ | $0.72_{\pm0.06}$ | $1.30_{\pm0.09}$ | $0.93_{\pm0.07}$ | $0.85_{\pm0.07}$ | $0.26_{\pm0.14}$ | $0.72_{\pm0.07}$ | $1.31_{\pm0.12}$ | $0.93_{\pm0.09}$ | $0.73_{\pm0.07}$ |
| CMAES | $-0.37_{\pm0.11}$ | $0.51_{\pm0.11}$ | $0.51_{\pm0.11}$ | $0.51_{\pm0.11}$ | $0.51_{\pm0.11}$ | - | | | | | - | | | | |
| TTDDEA | $-0.40_{\pm0.13}$ | $-0.47_{\pm0.28}$ | $-0.47_{\pm0.28}$ | $-0.47_{\pm0.28}$ | $-0.47_{\pm0.28}$ | $-0.41_{\pm0.14}$ | $-0.29_{\pm0.27}$ | $-0.29_{\pm0.27}$ | $-0.29_{\pm0.27}$ | $-0.29_{\pm0.27}$ | $-0.34_{\pm0.13}$ | $-0.21_{\pm0.27}$ | $-0.21_{\pm0.27}$ | $-0.21_{\pm0.27}$ | $-0.21_{\pm0.27}$ |
| Trimentoring | $-0.21_{\pm0.09}$ | $0.32_{\pm0.20}$ | $0.32_{\pm0.20}$ | $0.32_{\pm0.20}$ | $0.32_{\pm0.20}$ | $-0.12_{\pm0.10}$ | $0.54_{\pm0.12}$ | $0.55_{\pm0.13}$ | $0.54_{\pm0.12}$ | $0.54_{\pm0.12}$ | $-0.11_{\pm0.11}$ | $0.64_{\pm0.10}$ | $0.64_{\pm0.11}$ | $0.64_{\pm0.10}$ | $0.64_{\pm0.10}$ |

Table 62: Overall results for mujoco unconstrained tasks with 128 solutions and 0th percentile evaluations. In this case, 25% of the values are missing near the worst value and another 25% near the optimal value. Details are the same as Table 5.

| Steps | t = 50 | | | | | t = 100 | | | | | t = 150 | | | | |
|---|---|---|---|---|---|---|---|---|---|---|---|---|---|---|---|
| **Task** $f(x^*_{OFF})$ | Mujoco 1  $0.47_{\pm0.00}$ | | | | | | | | | | | | | | |
| Metric | FS | SI | OI | SO | $SO_\omega$ | FS | SI | OI | SO | $SO_\omega$ | FS | SI | OI | SO | $SO_\omega$ |
| ARCOO | $0.35_{\pm0.11}$ | $0.93_{\pm0.09}$ | $0.93_{\pm0.09}$ | $0.93_{\pm0.09}$ | $0.93_{\pm0.09}$ | $0.17_{\pm0.04}$ | $0.64_{\pm0.16}$ | $0.64_{\pm0.16}$ | $0.64_{\pm0.16}$ | $0.64_{\pm0.16}$ | $0.17_{\pm0.05}$ | $0.49_{\pm0.16}$ | $0.49_{\pm0.16}$ | $0.49_{\pm0.16}$ | $0.49_{\pm0.16}$ |
| BO | $0.15_{\pm0.03}$ | $0.37_{\pm0.02}$ | $0.37_{\pm0.02}$ | $0.37_{\pm0.02}$ | $0.37_{\pm0.02}$ | $0.15_{\pm0.03}$ | $0.37_{\pm0.02}$ | $0.37_{\pm0.02}$ | $0.37_{\pm0.02}$ | $0.37_{\pm0.02}$ | $0.14_{\pm0.04}$ | $0.36_{\pm0.01}$ | $0.36_{\pm0.01}$ | $0.36_{\pm0.01}$ | $0.36_{\pm0.01}$ |
| CBAS | $0.00_{\pm0.00}$ | $-45.62_{\pm0.25}$ | $-45.62_{\pm0.25}$ | $-45.62_{\pm0.25}$ | $-45.62_{\pm0.25}$ | $0.00_{\pm0.00}$ | $-45.86_{\pm0.26}$ | $-45.86_{\pm0.26}$ | $-45.86_{\pm0.26}$ | $-45.86_{\pm0.26}$ | $0.00_{\pm0.00}$ | $-45.94_{\pm0.26}$ | $-45.94_{\pm0.26}$ | $-45.94_{\pm0.26}$ | $-45.94_{\pm0.26}$ |
| CCDDEA | $0.69_{\pm0.17}$ | $0.64_{\pm0.08}$ | $1.24_{\pm0.19}$ | $0.85_{\pm0.11}$ | $0.95_{\pm0.13}$ | $0.73_{\pm0.13}$ | $0.74_{\pm0.09}$ | $1.43_{\pm0.21}$ | $0.98_{\pm0.12}$ | $0.89_{\pm0.11}$ | $0.73_{\pm0.13}$ | $0.78_{\pm0.10}$ | $1.49_{\pm0.23}$ | $1.02_{\pm0.14}$ | $0.78_{\pm0.10}$ |
| CMAES | $0.27_{\pm0.04}$ | $0.56_{\pm0.07}$ | $0.56_{\pm0.07}$ | $0.56_{\pm0.07}$ | $0.56_{\pm0.07}$ | - | | | | | - | | | | |
| TTDDEA | $0.25_{\pm0.12}$ | $-0.33_{\pm0.56}$ | $-0.33_{\pm0.56}$ | $-0.33_{\pm0.56}$ | $-0.33_{\pm0.56}$ | $0.23_{\pm0.10}$ | $-0.14_{\pm0.47}$ | $-0.14_{\pm0.47}$ | $-0.14_{\pm0.47}$ | $-0.14_{\pm0.47}$ | $0.24_{\pm0.07}$ | $-0.08_{\pm0.44}$ | $-0.08_{\pm0.44}$ | $-0.08_{\pm0.44}$ | $-0.08_{\pm0.44}$ |
| Trimentoring | $0.32_{\pm0.05}$ | $0.61_{\pm0.07}$ | $0.61_{\pm0.07}$ | $0.61_{\pm0.07}$ | $0.61_{\pm0.07}$ | $0.38_{\pm0.08}$ | $0.68_{\pm0.07}$ | $0.68_{\pm0.07}$ | $0.68_{\pm0.07}$ | $0.68_{\pm0.07}$ | $0.38_{\pm0.07}$ | $0.72_{\pm0.08}$ | $0.73_{\pm0.09}$ | $0.73_{\pm0.08}$ | $0.72_{\pm0.08}$ |
| **Task** $f(x^*_{OFF})$ | Mujoco 2  $0.07_{\pm0.00}$ | | | | | | | | | | | | | | |
| ARCOO | $0.01_{\pm0.13}$ | $0.97_{\pm0.01}$ | $0.97_{\pm0.01}$ | $0.97_{\pm0.01}$ | $0.97_{\pm0.01}$ | $-0.77_{\pm0.51}$ | $0.42_{\pm0.23}$ | $0.42_{\pm0.23}$ | $0.42_{\pm0.23}$ | $0.42_{\pm0.23}$ | $-0.82_{\pm0.22}$ | $-0.11_{\pm0.54}$ | $-0.11_{\pm0.54}$ | $-0.11_{\pm0.54}$ | $-0.11_{\pm0.54}$ |
| BO | $-0.92_{\pm0.41}$ | $0.22_{\pm0.27}$ | $0.22_{\pm0.27}$ | $0.22_{\pm0.27}$ | $0.22_{\pm0.27}$ | $-0.95_{\pm0.42}$ | $0.20_{\pm0.27}$ | $0.20_{\pm0.27}$ | $0.20_{\pm0.27}$ | $0.20_{\pm0.27}$ | $-0.93_{\pm0.41}$ | $0.19_{\pm0.27}$ | $0.19_{\pm0.27}$ | $0.19_{\pm0.27}$ | $0.19_{\pm0.27}$ |
| CBAS | $-1.86_{\pm0.00}$ | $-189.94_{\pm0.32}$ | $-189.94_{\pm0.32}$ | $-189.94_{\pm0.32}$ | $-189.94_{\pm0.32}$ | $-1.86_{\pm0.00}$ | $-190.93_{\pm0.33}$ | $-190.93_{\pm0.33}$ | $-190.93_{\pm0.33}$ | $-190.93_{\pm0.33}$ | $-1.86_{\pm0.00}$ | $-191.26_{\pm0.33}$ | $-191.26_{\pm0.33}$ | $-191.26_{\pm0.33}$ | $-191.26_{\pm0.33}$ |
| CCDDEA | $0.08_{\pm0.25}$ | $0.70_{\pm0.09}$ | $1.25_{\pm0.18}$ | $0.89_{\pm0.12}$ | $0.99_{\pm0.13}$ | $0.33_{\pm0.18}$ | $0.72_{\pm0.09}$ | $1.30_{\pm0.20}$ | $0.93_{\pm0.12}$ | $0.85_{\pm0.11}$ | $0.33_{\pm0.18}$ | $0.74_{\pm0.11}$ | $1.34_{\pm0.23}$ | $0.96_{\pm0.14}$ | $0.75_{\pm0.11}$ |
| CMAES | $-0.40_{\pm0.09}$ | $0.47_{\pm0.07}$ | $0.47_{\pm0.07}$ | $0.47_{\pm0.07}$ | $0.47_{\pm0.07}$ | - | | | | | - | | | | |
| TTDDEA | $-0.30_{\pm0.17}$ | $-0.47_{\pm0.49}$ | $-0.47_{\pm0.49}$ | $-0.47_{\pm0.49}$ | $-0.47_{\pm0.49}$ | $-0.33_{\pm0.12}$ | $-0.28_{\pm0.46}$ | $-0.28_{\pm0.46}$ | $-0.28_{\pm0.46}$ | $-0.28_{\pm0.46}$ | $-0.31_{\pm0.14}$ | $-0.22_{\pm0.46}$ | $-0.22_{\pm0.46}$ | $-0.22_{\pm0.46}$ | $-0.22_{\pm0.46}$ |
| Trimentoring | $-0.21_{\pm0.13}$ | $0.52_{\pm0.06}$ | $0.52_{\pm0.06}$ | $0.52_{\pm0.06}$ | $0.52_{\pm0.06}$ | $-0.09_{\pm0.12}$ | $0.65_{\pm0.07}$ | $0.65_{\pm0.08}$ | $0.65_{\pm0.07}$ | $0.65_{\pm0.07}$ | $-0.07_{\pm0.12}$ | $0.70_{\pm0.08}$ | $0.71_{\pm0.09}$ | $0.71_{\pm0.08}$ | $0.70_{\pm0.08}$ |

Table 63: Overall results for mujoco unconstrained tasks with 128 solutions and 0th percentile evaluations. In this case, 30% of the values are missing near the worst value and another 20% near the optimal value. Details are the same as Table 5.

| Steps | | t = 50 | | | | | t = 100 | | | | | t = 150 | | | |
|---|---|---|---|---|---|---|---|---|---|---|---|---|---|---|---|
| Task $f(x^*_{OFF})$ | | | | | | | Mujoco 1  0.5±0.00 | | | | | | | | |
| Metric | FS | SI | OI | SO | SO_ω | FS | SI | OI | SO | SO_ω | FS | SI | OI | SO | SO_ω |
| ARCOO | 0.41±0.12 | 0.94±0.07 | 0.94±0.07 | 0.94±0.07 | 0.94±0.07 | 0.15±0.03 | 0.59±0.08 | 0.59±0.08 | 0.59±0.08 | 0.59±0.08 | 0.15±0.05 | 0.41±0.06 | 0.41±0.06 | 0.41±0.06 | 0.41±0.06 |
| BO | 0.15±0.05 | 0.26±0.02 | 0.26±0.02 | 0.26±0.02 | 0.26±0.02 | 0.14±0.03 | 0.26±0.02 | 0.26±0.02 | 0.26±0.02 | 0.26±0.02 | 0.15±0.03 | 0.26±0.02 | 0.26±0.02 | 0.26±0.02 | 0.26±0.02 |
| CBAS | 0.00±0.00 | -48.64±0.34 | -48.64±0.34 | -48.64±0.34 | -48.64±0.34 | 0.00±0.00 | -48.89±0.34 | -48.89±0.34 | -48.89±0.34 | -48.89±0.34 | 0.00±0.00 | -48.98±0.34 | -48.98±0.34 | -48.98±0.34 | -48.98±0.34 |
| CCDDEA | 0.73±0.14 | 0.65±0.08 | 1.22±0.20 | 0.85±0.11 | 0.95±0.13 | 0.73±0.14 | 0.76±0.10 | 1.42±0.25 | 0.99±0.14 | 0.90±0.12 | 0.73±0.14 | 0.79±0.11 | 1.48±0.28 | 1.03±0.16 | 0.79±0.11 |
| CMAES | 0.32±0.04 | 0.54±0.06 | 0.54±0.06 | 0.54±0.06 | 0.54±0.06 | - | - | - | - | - | - | - | - | - | - |
| TTDDEA | 0.23±0.08 | -0.85±0.63 | -0.85±0.63 | -0.85±0.63 | -0.85±0.63 | 0.24±0.09 | -0.61±0.51 | -0.61±0.51 | -0.61±0.51 | -0.61±0.51 | 0.29±0.00 | -0.52±0.47 | -0.52±0.47 | -0.52±0.47 | -0.52±0.47 |
| Trimentoring | 0.32±0.06 | 0.52±0.11 | 0.52±0.11 | 0.52±0.11 | 0.52±0.11 | 0.35±0.06 | 0.57±0.09 | 0.57±0.09 | 0.57±0.09 | 0.57±0.09 | 0.38±0.06 | 0.62±0.07 | 0.62±0.07 | 0.62±0.07 | 0.62±0.07 |
| Task $f(x^*_{OFF})$ | | | | | | | Mujoco 2  0.11±0.00 | | | | | | | | |
| ARCOO | 0.09±0.03 | 0.98±0.01 | 0.98±0.01 | 0.98±0.01 | 0.98±0.01 | -0.57±0.21 | 0.68±0.14 | 0.68±0.14 | 0.68±0.14 | 0.68±0.14 | -0.66±0.18 | 0.34±0.17 | 0.34±0.17 | 0.34±0.17 | 0.34±0.17 |
| BO | -0.92±0.55 | 0.08±0.26 | 0.08±0.26 | 0.08±0.26 | 0.08±0.26 | -1.14±0.58 | 0.07±0.26 | 0.07±0.26 | 0.07±0.26 | 0.07±0.26 | -1.86±0.00 | 0.07±0.26 | 0.07±0.26 | 0.07±0.26 | 0.07±0.26 |
| CBAS | -1.86±0.00 | -194.21±0.35 | -194.21±0.35 | -194.21±0.35 | -194.21±0.35 | -1.86±0.00 | -195.23±0.35 | -195.23±0.35 | -195.23±0.35 | -195.23±0.35 | -1.86±0.00 | -195.56±0.35 | -195.56±0.35 | -195.56±0.35 | -195.56±0.35 |
| CCDDEA | 0.18±0.21 | 0.65±0.07 | 1.16±0.13 | 0.83±0.09 | 0.92±0.10 | 0.29±0.11 | 0.65±0.05 | 1.20±0.07 | 0.84±0.05 | 0.77±0.05 | 0.29±0.11 | 0.66±0.06 | 1.21±0.07 | 0.85±0.06 | 0.66±0.06 |
| CMAES | -0.36±0.12 | 0.40±0.12 | 0.40±0.12 | 0.40±0.12 | 0.40±0.12 | - | - | - | - | - | - | - | - | - | - |
| TTDDEA | -0.35±0.13 | -0.73±0.45 | -0.73±0.45 | -0.73±0.45 | -0.73±0.45 | -0.33±0.13 | -0.51±0.35 | -0.51±0.35 | -0.51±0.35 | -0.51±0.35 | -0.39±0.09 | -0.43±0.33 | -0.43±0.33 | -0.43±0.33 | -0.43±0.33 |
| Trimentoring | -0.26±0.37 | 0.42±0.18 | 0.43±0.19 | 0.43±0.18 | 0.43±0.18 | -0.15±0.20 | 0.56±0.16 | 0.58±0.14 | 0.57±0.17 | 0.57±0.17 | -0.10±0.21 | 0.61±0.18 | 0.63±0.21 | 0.62±0.19 | 0.61±0.18 |

Table 64: Overall results for mujoco unconstrained tasks with 128 solutions and 0th percentile evaluations. In this case, 40% of the values are missing near the worst value and another 10% near the optimal value. Details are the same as Table 5.

| Steps | | t = 50 | | | | | t = 100 | | | | | t = 150 | | | |
|---|---|---|---|---|---|---|---|---|---|---|---|---|---|---|---|
| Task $f(x^*_{OFF})$ | | | | | | | Mujoco 1  0.59±0.01 | | | | | | | | |
| Metric | FS | SI | OI | SO | SO_ω | FS | SI | OI | SO | SO_ω | FS | SI | OI | SO | SO_ω |
| ARCOO | 0.43±0.12 | 0.90±0.04 | 0.90±0.04 | 0.90±0.04 | 0.90±0.04 | 0.23±0.06 | 0.47±0.09 | 0.47±0.09 | 0.47±0.09 | 0.47±0.09 | 0.19±0.06 | 0.23±0.09 | 0.23±0.09 | 0.23±0.09 | 0.23±0.09 |
| BO | 0.15±0.05 | -0.13±0.04 | -0.13±0.04 | -0.13±0.04 | -0.13±0.04 | 0.15±0.03 | -0.13±0.04 | -0.13±0.04 | -0.13±0.04 | -0.13±0.04 | 0.15±0.03 | -0.14±0.04 | -0.14±0.04 | -0.14±0.04 | -0.14±0.04 |
| CBAS | 0.00±0.00 | -57.79±0.58 | -57.79±0.58 | -57.79±0.58 | -57.79±0.58 | 0.00±0.00 | -58.10±0.58 | -58.10±0.58 | -58.10±0.58 | -58.10±0.58 | 0.00±0.00 | -58.20±0.59 | -58.20±0.59 | -58.20±0.59 | -58.20±0.59 |
| CCDDEA | 0.74±0.11 | 0.54±0.21 | 0.98±0.41 | 0.69±0.28 | 0.77±0.31 | 0.76±0.06 | 0.71±0.10 | 1.26±0.23 | 0.90±0.14 | 0.82±0.12 | 0.76±0.06 | 0.76±0.10 | 1.35±0.19 | 0.97±0.11 | 0.76±0.10 |
| CMAES | 0.39±0.03 | 0.31±0.08 | 0.31±0.08 | 0.31±0.08 | 0.31±0.08 | - | - | - | - | - | - | - | - | - | - |
| TTDDEA | 0.14±0.07 | -6.50±7.04 | -6.50±7.04 | -6.50±7.04 | -6.50±7.04 | 0.20±0.11 | -6.25±6.89 | -6.25±6.89 | -6.25±6.89 | -6.25±6.89 | 0.14±0.08 | -5.98±6.59 | -5.98±6.59 | -5.98±6.59 | -5.98±6.59 |
| Trimentoring | 0.38±0.11 | 0.30±0.12 | 0.30±0.13 | 0.30±0.12 | 0.30±0.12 | 0.41±0.10 | 0.40±0.16 | 0.41±0.18 | 0.41±0.18 | 0.40±0.17 | 0.46±0.09 | 0.47±0.16 | 0.48±0.19 | 0.48±0.17 | 0.47±0.16 |
| Task $f(x^*_{OFF})$ | | | | | | | Mujoco 2  0.23±0.01 | | | | | | | | |
| ARCOO | 0.21±0.01 | 0.94±0.03 | 0.94±0.03 | 0.94±0.03 | 0.94±0.03 | -0.49±0.15 | 0.28±0.36 | 0.28±0.36 | 0.28±0.36 | 0.28±0.36 | -0.97±0.54 | -0.56±0.83 | -0.56±0.83 | -0.56±0.83 | -0.56±0.83 |
| BO | -0.89±0.40 | -0.02±0.36 | -0.02±0.36 | -0.02±0.36 | -0.02±0.36 | -0.71±0.21 | -0.04±0.35 | -0.04±0.35 | -0.04±0.35 | -0.04±0.35 | -0.69±0.14 | -0.06±0.35 | -0.06±0.35 | -0.06±0.35 | -0.06±0.35 |
| CBAS | -1.86±0.00 | -205.86±0.46 | -205.86±0.46 | -205.86±0.46 | -205.86±0.46 | -1.86±0.00 | -206.94±0.46 | -206.94±0.46 | -206.94±0.46 | -206.94±0.46 | -1.86±0.00 | -207.29±0.46 | -207.29±0.46 | -207.29±0.46 | -207.29±0.46 |
| CCDDEA | 0.17±0.20 | 0.57±0.11 | 0.98±0.19 | 0.72±0.13 | 0.79±0.14 | 0.27±0.13 | 0.57±0.09 | 1.02±0.16 | 0.73±0.10 | 0.67±0.10 | 0.27±0.13 | 0.57±0.11 | 1.02±0.17 | 0.73±0.12 | 0.57±0.11 |
| CMAES | -0.31±0.15 | 0.20±0.15 | 0.20±0.15 | 0.20±0.15 | 0.20±0.15 | - | - | - | - | - | - | - | - | - | - |
| TTDDEA | -0.38±0.12 | -10.97±18.12 | -10.97±18.12 | -10.97±18.12 | -10.97±18.12 | -0.34±0.30 | -10.11±17.05 | -10.11±17.05 | -10.11±17.05 | -10.11±17.05 | -0.28±0.01 | -9.92±16.97 | -9.92±16.97 | -9.92±16.97 | -9.92±16.97 |
| Trimentoring | -0.25±0.25 | 0.17±0.34 | 0.17±0.34 | 0.17±0.34 | 0.17±0.34 | -0.24±0.24 | 0.27±0.34 | 0.27±0.34 | 0.27±0.34 | 0.27±0.34 | -0.14±0.10 | 0.32±0.36 | 0.32±0.36 | 0.32±0.36 | 0.32±0.36 |

Table 65: Overall results for CEC constrained tasks with 128 solutions and 100th percentile evaluations. In this case, 0% of the values are missing near the worst value and another 50% near the optimal value. Details are the same as Table 5.

| Steps | | t = 50 | | | | | t = 100 | | | | | t = 150 | | | |
|---|---|---|---|---|---|---|---|---|---|---|---|---|---|---|---|
| Task $f(x^*_{OFF})$ | | | | | | | CEC 1  -2.08e4±424.97 | | | | | | | | |
| Metric | FS | SI | OI | SO | SO_ω | FS | SI | OI | SO | SO_ω | FS | SI | OI | SO | SO_ω |
| CARCOO | - | - | - | - | - | - | - | - | - | - | - | - | - | - | - |
| DEPF | -2.08e4±424.97 | 1.00±0.00 | 1.00±0.00 | 1.00±0.00 | 1.00±0.00 | -2.08e4±424.97 | 1.00±0.00 | 1.00±0.00 | 1.00±0.00 | 1.00±0.00 | -2.08e4±424.97 | 1.00±0.00 | 1.00±0.00 | 1.00±0.00 | 1.00±0.00 |
| DESPF | -2.08e4±424.97 | 1.00±0.00 | 1.00±0.00 | 1.00±0.00 | 1.00±0.00 | -2.08e4±424.97 | 1.00±0.00 | 1.00±0.00 | 1.00±0.00 | 1.00±0.00 | -2.08e4±424.97 | 1.00±0.00 | 1.00±0.00 | 1.00±0.00 | 1.00±0.00 |
| PRIME | - | - | - | - | - | - | - | - | - | - | - | - | - | - | - |
| Task $f(x^*_{OFF})$ | | | | | | | CEC 2  -1.14±0.01 | | | | | | | | |
| CARCOO | -2.05±0.00 | -66.42±35.86 | -69.31±32.18 | -67.38±34.55 | -67.87±33.92 | -2.05±0.00 | -67.20±35.29 | -70.55±30.66 | -68.32±33.68 | -67.88±34.30 | -2.05±0.00 | -67.46±35.10 | -70.95±30.56 | -68.62±33.40 | -67.47±35.08 |
| DEPF | -1.25±0.00 | -9.81±3.29 | -10.02±2.78 | -9.88±3.12 | -9.92±3.03 | -1.25±0.00 | -9.51±3.09 | -9.71±2.56 | -9.57±2.92 | -9.55±2.99 | -1.35±0.26 | -11.96±7.76 | -12.15±7.51 | -12.02±7.67 | -11.96±7.76 |
| DESPF | -1.47±0.33 | -17.61±6.78 | -17.61±6.78 | -17.61±6.78 | -17.61±6.78 | -1.57±0.37 | -28.15±19.23 | -28.15±19.23 | -28.15±19.23 | -28.15±19.23 | -1.49±0.33 | -31.15±23.35 | -31.15±23.35 | -31.15±23.35 | -31.15±23.35 |
| PRIME | -2.04±0.00 | -3.50±0.64 | -6.99±1.29 | -4.66±0.86 | -5.25±0.97 | -2.04±0.00 | -6.01±0.79 | -12.02±1.57 | -8.01±1.08 | -7.23±0.94 | -2.04±0.00 | -6.83±0.88 | -13.67±1.76 | -9.11±1.18 | -6.86±0.89 |
| Task $f(x^*_{OFF})$ | | | | | | | CEC 3  -2.35±0.02 | | | | | | | | |
| CARCOO | -3.23±0.97 | -1.48±2.73 | -2.95±5.46 | -1.97±3.64 | -2.22±4.10 | -3.47±0.94 | -2.66±2.95 | -5.32±5.90 | -3.55±3.93 | -3.20±3.55 | -3.48±0.95 | -3.06±3.14 | -6.12±6.29 | -4.08±4.19 | -3.07±3.15 |
| DEPF | -3.28±0.98 | -0.09±0.90 | -0.18±1.80 | -0.12±1.20 | -0.14±1.35 | -3.41±0.89 | -0.61±1.11 | -1.23±2.22 | -0.82±1.48 | -0.74±1.35 | -3.41±0.90 | -0.80±1.18 | -1.59±2.37 | -1.06±1.58 | -0.80±1.19 |
| DESPF | -3.16±0.95 | -0.38±0.86 | -0.76±1.73 | -0.50±1.15 | -0.57±1.30 | -3.43±0.70 | -0.80±0.95 | -1.60±1.90 | -1.06±1.27 | -0.80±0.95 | -3.43±0.70 | -0.80±0.95 | -1.60±1.90 | -1.06±1.27 | -0.80±0.95 |
| PRIME | -3.40±0.82 | -1.36±0.46 | -2.71±0.91 | -1.81±0.61 | -2.04±0.69 | -4.20±0.00 | -1.54±0.32 | -3.07±0.63 | -2.05±0.42 | -1.85±0.38 | -4.20±0.00 | -1.75±0.25 | -3.49±0.50 | -2.33±0.33 | -1.75±0.25 |
| Task $f(x^*_{OFF})$ | | | | | | | CEC 4  -264.43±0.30 | | | | | | | | |
| CARCOO | -379.59±1.78 | -1.14e4±195.53 | -1.14e4±195.53 | -1.14e4±195.53 | -1.14e4±195.53 | -379.59±1.78 | -1.15e4±196.54 | -1.15e4±196.54 | -1.15e4±196.54 | -1.15e4±196.54 | -379.59±1.78 | -1.15e4±196.88 | -1.15e4±196.88 | -1.15e4±196.88 | -1.15e4±196.88 |
| DEPF | -299.92±21.31 | -2.14e3±548.32 | -2.14e3±548.32 | -2.14e3±548.32 | -2.14e3±548.32 | -313.88±17.33 | -3.09e3±705.87 | -3.09e3±705.87 | -3.09e3±705.87 | -3.09e3±705.87 | -323.13±17.69 | -3.76e3±818.66 | -3.76e3±818.66 | -3.76e3±818.66 | -3.76e3±818.66 |
| DESPF | -305.65±17.02 | -2.11e3±394.73 | -2.11e3±394.73 | -2.11e3±394.73 | -2.11e3±394.73 | -306.97±19.29 | -2.86e3±780.18 | -2.86e3±780.18 | -2.86e3±780.18 | -2.86e3±780.18 | -319.22±19.25 | -3.50e3±1.16e3 | -3.50e3±1.16e3 | -3.50e3±1.16e3 | -3.50e3±1.16e3 |
| PRIME | -379.59±1.78 | -1.14e4±195.53 | -1.14e4±195.53 | -1.14e4±195.53 | -1.14e4±195.53 | -379.59±1.78 | -1.14e4±215.30 | -1.14e4±215.30 | -1.14e4±215.30 | -1.14e4±215.30 | -379.59±1.78 | -1.14e4±201.24 | -1.14e4±201.24 | -1.14e4±201.24 | -1.14e4±201.24 |
| Task $f(x^*_{OFF})$ | | | | | | | CEC 5  -10.71±0.18 | | | | | | | | |
| CARCOO | -58.62±3.36 | -1.91±0.18 | -3.81±0.37 | -2.54±0.24 | -2.87±0.28 | -58.62±3.36 | -2.15±0.18 | -4.31±0.37 | -2.87±0.24 | -2.59±0.22 | -58.62±3.36 | -2.23±0.18 | -4.47±0.37 | -2.98±0.24 | -2.24±0.18 |
| DEPF | -16.16±2.17 | 0.51±0.04 | 1.03±0.08 | 0.69±0.05 | 0.77±0.06 | -22.79±5.96 | 0.16±0.19 | 0.32±0.38 | 0.21±0.26 | 0.19±0.25 | -24.51±9.34 | -0.03±0.30 | -0.07±0.59 | -0.04±0.40 | -0.03±0.30 |
| DESPF | -40.50±16.88 | -0.69±0.88 | -1.38±1.79 | -0.92±1.17 | -1.04±1.31 | -50.76±14.09 | -1.87±1.19 | -3.73±2.38 | -2.49±1.59 | -2.25±1.43 | -53.50±14.11 | -2.30±1.31 | -4.60±2.62 | -3.07±1.75 | -2.31±1.31 |
| PRIME | -58.62±3.36 | -2.10±0.23 | -4.21±0.47 | -2.81±0.31 | -3.16±0.35 | -58.62±3.36 | -2.40±0.23 | -4.80±0.47 | -3.20±0.31 | -2.89±0.28 | -58.62±3.36 | -2.50±0.24 | -5.00±0.47 | -3.33±0.32 | -2.51±0.24 |

Table 66: Overall results for CEC constrained tasks with 128 solutions and 100th percentile evaluations. In this case, 0% of the values are missing near the worst value and another 40% near the optimal value. Details are the same as Table 5.

| Steps | t = 50 | | | | | t = 100 | | | | | t = 150 | | | | |
|---|---|---|---|---|---|---|---|---|---|---|---|---|---|---|---|
| **Task** $f(x^*_{OFF})$ | CEC 1 −2.08e4±424.97 | | | | | | | | | | | | | | |
| Metric | FS | SI | OI | SO | SO_w | FS | SI | OI | SO | SO_w | FS | SI | OI | SO | SO_w |
| CARCOO | - | - | - | - | - | - | - | - | - | - | - | - | - | - | - |
| DEPF | −2.08e4±424.97 | 1.00±.00 | 1.00±.00 | 1.00±.00 | 1.00±.00 | −2.08e4±424.97 | 1.00±.00 | 1.00±.00 | 1.00±.00 | 1.00±.00 | −2.08e4±424.97 | 1.00±.00 | 1.00±.00 | 1.00±.00 | 1.00±.00 |
| DESPF | −2.08e4±424.97 | 1.00±.00 | 1.00±.00 | 1.00±.00 | 1.00±.00 | −2.08e4±424.97 | 1.00±.00 | 1.00±.00 | 1.00±.00 | 1.00±.00 | −2.08e4±424.97 | 1.00±.00 | 1.00±.00 | 1.00±.00 | 1.00±.00 |
| PRIME | - | - | - | - | - | - | - | - | - | - | - | - | - | - | - |
| **Task** $f(x^*_{OFF})$ | CEC 2 −1.08±0.00 | | | | | | | | | | | | | | |
| CARCOO | −1.81±.40 | −73.36±36.20 | −74.24±34.91 | −73.73±35.64 | −73.88±35.42 | −1.81±.40 | −72.63±38.18 | −73.07±37.66 | −72.81±37.88 | −72.75±37.99 | −1.81±.41 | −73.58±36.90 | −74.35±35.74 | −73.91±36.41 | −73.59±36.90 |
| DEPF | −1.26±.03 | −17.73±1.62 | −17.73±1.62 | −17.73±1.62 | −17.73±1.62 | −1.45±.34 | −23.51±11.62 | −23.51±11.62 | −23.51±11.62 | −23.51±11.62 | −1.46±.34 | −27.84±19.12 | −27.84±19.12 | −27.84±19.12 | −27.84±19.12 |
| DESPF | −1.39±.25 | −24.44±8.24 | −24.44±8.24 | −24.44±8.24 | −24.44±8.24 | −1.56±.38 | −30.02±16.84 | −30.02±16.84 | −30.02±16.84 | −30.02±16.84 | −1.56±.38 | −35.06±22.30 | −35.06±22.30 | −35.06±22.30 | −35.06±22.30 |
| PRIME | −2.04±.00 | −89.69±1.91 | −89.69±1.91 | −89.69±1.91 | −89.69±1.91 | −2.04±.00 | −92.50±1.02 | −92.50±1.02 | −92.50±1.02 | −92.50±1.02 | −2.04±.00 | −93.43±.79 | −93.43±.79 | −93.43±.79 | −93.43±.79 |
| **Task** $f(x^*_{OFF})$ | CEC 3 −2.12±0.02 | | | | | | | | | | | | | | |
| CARCOO | −4.20±.00 | −8.78±1.35 | −17.56±2.70 | −11.71±1.80 | −13.20±2.02 | −4.20±.00 | −9.03±1.35 | −18.05±2.69 | −12.03±1.80 | −10.86±1.62 | −4.20±.00 | −9.10±1.35 | −18.21±2.70 | −12.14±1.80 | −9.13±1.35 |
| DEPF | −3.31±.34 | −3.02±1.57 | −6.03±3.15 | −4.02±2.10 | −4.53±2.36 | −3.56±.49 | −4.27±1.66 | −8.55±3.33 | −5.70±2.22 | −5.14±2.00 | −3.56±.49 | −4.87±1.82 | −9.74±3.65 | −6.49±2.43 | −4.88±1.83 |
| DESPF | −3.44±.68 | −32.09±52.45 | −35.29±51.34 | −33.20±52.10 | −33.74±51.91 | −3.70±.50 | −38.83±60.48 | −43.96±59.60 | −40.59±59.85 | −39.91±60.10 | −3.70±.50 | −41.18±63.05 | −47.07±60.88 | −43.19±62.32 | −41.20±63.04 |
| PRIME | −3.34±.87 | −5.09±1.47 | −10.18±2.94 | −6.79±1.96 | −7.65±2.21 | −4.00±.52 | −6.13±.97 | −12.26±1.95 | −8.17±1.30 | −7.38±1.17 | −4.20±.00 | −6.88±.89 | −13.76±1.77 | −9.17±1.18 | −6.90±.89 |
| **Task** $f(x^*_{OFF})$ | CEC 4 −264.43±0.30 | | | | | | | | | | | | | | |
| CARCOO | −379.59±1.78 | −1.14e4±195.53 | −1.14e4±195.53 | −1.14e4±195.53 | −1.14e4±195.53 | −379.59±1.78 | −1.15e4±196.54 | −1.15e4±196.54 | −1.15e4±196.54 | −1.15e4±196.54 | −379.59±1.78 | −1.15e4±196.88 | −1.15e4±196.88 | −1.15e4±196.88 | −1.15e4±196.88 |
| DEPF | −302.43±13.86 | −1.58e3±914.04 | −1.59e3±891.19 | −1.58e3±906.41 | −1.58e3±902.53 | −320.39±22.51 | −2.74e3±1.1bs3 | −2.76e3±1.19bs3 | −2.74e3±1.63bs3 | −2.74e3±1.62bs3 | −327.34±24.05 | −3.30e3±1.97bs3 | −3.33e3±1.93bs3 | −3.31e3±1.96bs3 | −3.30e3±1.97bs3 |
| DESPF | −298.22±10.40 | −1.70e3±700.12 | −1.71e3±679.71 | −1.70e3±693.29 | −1.71e3±689.84 | −2.96e3±1.27e3 | −2.97e3±1.24e3 | −2.96e3±1.26e3 | −2.96e3±1.27e3 | −325.76±19.05 | −3.73e3±1.66e3 | −3.75e3±1.65e3 | −3.74e3±1.65e3 | −3.73e3±1.66e3 | |
| PRIME | −379.59±1.78 | −1.12e4±447.48 | −1.12e4±447.48 | −1.12e4±447.48 | −1.12e4±447.48 | −366.28±34.89 | −1.07e4±1.25e3 | −1.07e4±1.25e3 | −1.07e4±1.25e3 | −1.07e4±1.25e3 | −379.59±1.78 | −1.06e4±1.02e3 | −1.06e4±1.02e3 | −1.06e4±1.02e3 | −1.06e4±1.02e3 |
| **Task** $f(x^*_{OFF})$ | CEC 5 −8.61±0.14 | | | | | | | | | | | | | | |
| CARCOO | −58.62±3.36 | −3.02±.21 | −6.03±.43 | −4.02±.29 | −4.53±.32 | −58.62±3.36 | −3.28±.20 | −6.56±.40 | −4.37±.27 | −3.94±.25 | −58.62±3.36 | −3.37±.20 | −6.73±.40 | −4.49±.27 | −3.38±.20 |
| DEPF | −15.94±3.01 | 0.38±.19 | 0.75±.38 | 0.50±.25 | 0.57±.28 | −20.33±5.53 | −0.13±.19 | −0.26±.39 | −0.17±.26 | −0.15±.23 | −21.89±5.71 | −0.37±.23 | −0.74±.46 | −0.49±.31 | −0.37±.23 |
| DESPF | −32.20±13.08 | −0.66±.38 | −1.32±.75 | −0.88±.50 | −1.00±.56 | −41.70±15.77 | −1.92±.95 | −3.84±1.91 | −2.56±1.27 | −2.31±1.15 | −47.94±14.87 | −2.46±1.05 | −4.93±2.10 | −3.28±1.40 | −2.47±1.06 |
| PRIME | −58.62±3.36 | −3.76±.35 | −7.53±.70 | −5.02±.47 | −5.65±.53 | −58.62±3.36 | −4.03±.35 | −8.07±.70 | −5.38±.47 | −4.85±.42 | −58.62±3.36 | −4.12±.35 | −8.25±.70 | −5.50±.47 | −4.14±.35 |

Table 67: Overall results for CEC constrained tasks with 128 solutions and 100th percentile evaluations. In this case, 0% of the values are missing near the worst value and another 30% near the optimal value. Details are the same as Table 5.

| Steps | t = 50 | | | | | t = 100 | | | | | t = 150 | | | | |
|---|---|---|---|---|---|---|---|---|---|---|---|---|---|---|---|
| **Task** $f(x^*_{OFF})$ | CEC 1 −2.08e4±424.97 | | | | | | | | | | | | | | |
| Metric | FS | SI | OI | SO | SO_w | FS | SI | OI | SO | SO_w | FS | SI | OI | SO | SO_w |
| CARCOO | - | - | - | - | - | - | - | - | - | - | - | - | - | - | - |
| DEPF | −2.08e4±424.97 | 1.00±.00 | 1.00±.00 | 1.00±.00 | 1.00±.00 | −2.08e4±424.97 | 1.00±.00 | 1.00±.00 | 1.00±.00 | 1.00±.00 | −2.08e4±424.97 | 1.00±.00 | 1.00±.00 | 1.00±.00 | 1.00±.00 |
| DESPF | −2.08e4±424.97 | 1.00±.00 | 1.00±.00 | 1.00±.00 | 1.00±.00 | −2.08e4±424.97 | 1.00±.00 | 1.00±.00 | 1.00±.00 | 1.00±.00 | −2.08e4±424.97 | 1.00±.00 | 1.00±.00 | 1.00±.00 | 1.00±.00 |
| PRIME | - | - | - | - | - | - | - | - | - | - | - | - | - | - | - |
| **Task** $f(x^*_{OFF})$ | CEC 2 −1.08±0.00 | | | | | | | | | | | | | | |
| CARCOO | −1.93±.29 | −83.25±29.02 | −83.25±29.02 | −83.25±29.02 | −83.25±29.02 | −1.93±.29 | −83.70±29.15 | −83.70±29.15 | −83.70±29.15 | −83.70±29.15 | −1.93±.29 | −83.85±29.18 | −83.85±29.18 | −83.85±29.18 | −83.85±29.18 |
| DEPF | −1.24±.03 | −17.13±1.39 | −17.13±1.39 | −17.13±1.39 | −17.13±1.39 | −1.27±.04 | −17.51±2.03 | −17.51±2.03 | −17.51±2.03 | −17.51±2.03 | −1.39±.25 | −20.79±7.39 | −20.79±7.39 | −20.79±7.39 | −20.79±7.39 |
| DESPF | −1.29±.05 | −20.16±4.08 | −20.16±4.08 | −20.16±4.08 | −20.16±4.08 | −1.40±.25 | −25.20±13.83 | −25.20±13.83 | −25.20±13.83 | −25.20±13.83 | −1.40±.25 | −27.07±17.58 | −27.07±17.58 | −27.07±17.58 | −27.07±17.58 |
| PRIME | −2.04±.00 | −91.01±2.97 | −91.01±2.97 | −91.01±2.97 | −91.01±2.97 | −2.04±.00 | −93.16±1.61 | −93.16±1.61 | −93.16±1.61 | −93.16±1.61 | −2.04±.00 | −93.86±1.20 | −93.86±1.20 | −93.86±1.20 | −93.86±1.20 |
| **Task** $f(x^*_{OFF})$ | CEC 3 −2.0±0.00 | | | | | | | | | | | | | | |
| CARCOO | −4.20±.00 | −182.71±14.80 | −195.24±17.78 | −188.68±15.76 | −190.82±16.31 | −4.20±.00 | −193.96±4.40 | −207.03±5.84 | −200.19±7.31 | −198.10±7.61 | −4.20±.00 | −197.66±7.51 | −210.91±5.91 | −203.97±5.30 | −197.74±7.47 |
| DEPF | −3.32±.60 | −88.07±27.29 | −88.18±27.18 | −88.12±27.14 | −88.14±27.14 | −3.45±.43 | −114.64±38.38 | −115.63±36.43 | −115.10±37.64 | −114.95±37.89 | −3.57±.49 | −124.70±39.68 | −126.01±37.96 | −125.31±38.83 | −124.71±39.67 |
| DESPF | −3.06±.33 | −64.79±31.79 | −64.79±31.79 | −64.79±31.79 | −64.79±31.79 | −3.44±.68 | −100.89±45.93 | −100.89±45.93 | −100.89±45.93 | −100.89±45.93 | −3.69±.50 | −119.62±50.89 | −119.62±50.89 | −119.62±50.89 | −119.62±50.89 |
| PRIME | −3.31±.92 | −196.06±25.51 | −196.06±25.51 | −196.06±25.51 | −196.06±25.51 | −4.04±.40 | −188.54±22.02 | −188.54±22.02 | −188.54±22.02 | −188.54±22.02 | −3.94±.60 | −194.62±14.02 | −194.62±14.02 | −194.62±14.02 | −194.62±14.02 |
| **Task** $f(x^*_{OFF})$ | CEC 4 −264.43±0.30 | | | | | | | | | | | | | | |
| CARCOO | −379.59±1.78 | −1.14e4±195.53 | −1.14e4±195.53 | −1.14e4±195.53 | −1.14e4±195.53 | −379.59±1.78 | −1.15e4±196.54 | −1.15e4±196.54 | −1.15e4±196.54 | −1.15e4±196.54 | −379.59±1.78 | −1.15e4±196.88 | −1.15e4±196.88 | −1.15e4±196.88 | −1.15e4±196.88 |
| DEPF | −300.43±13.32 | −1.76e3±699.59 | −1.76e3±688.98 | −1.76e3±694.25 | −1.76e3±694.25 | −310.41±14.62 | −2.77e3±1.15e3 | −2.78e3±1.12e3 | −2.77e3±1.14e3 | −2.77e3±1.14e3 | −336.14±16.28 | −3.62e3±1.41e3 | −3.64e3±1.37e3 | −3.62e3±1.40e3 | −3.62e3±1.41e3 |
| DESPF | −299.62±12.55 | −1.20e3±914.37 | −1.22e3±890.26 | −1.21e3±906.16 | −1.21e3±902.06 | −319.76±25.27 | −2.15e3±1.67e3 | −2.18e3±1.63e3 | −2.16e3±1.66e3 | −2.15e3±1.66e3 | −340.00±24.67 | −2.76e3±2.15e3 | −2.80e3±2.10e3 | −2.78e3±2.14e3 | −2.76e3±2.15e3 |
| PRIME | −339.05±21.93 | −9.14e3±3.58e3 | −9.14e3±3.58e3 | −9.14e3±3.58e3 | −9.14e3±3.58e3 | −356.86±40.34 | −8.35e3±4.11e3 | −8.38e3±4.06e3 | −8.36e3±4.09e3 | −8.36e3±4.10e3 | −379.59±1.78 | −8.19e3±3.69e3 | −8.22e3±3.65e3 | −8.20e3±3.67e3 | −8.19e3±3.69e3 |
| **Task** $f(x^*_{OFF})$ | CEC 5 −6.75±0.10 | | | | | | | | | | | | | | |
| CARCOO | −58.62±3.36 | −4.92±.30 | −9.84±.61 | −6.56±.41 | −7.39±.46 | −58.62±3.36 | −5.20±.31 | −10.40±.63 | −6.93±.42 | −6.26±.38 | −58.62±3.36 | −5.29±.32 | −10.58±.63 | −7.06±.42 | −5.31±.32 |
| DEPF | −15.33±4.08 | 0.09±.32 | 0.17±.64 | 0.12±.43 | 0.13±.48 | −25.03±6.09 | −0.82±.63 | −1.65±1.26 | −1.10±.84 | −0.99±.76 | −30.86±9.25 | −1.55±.97 | −3.10±1.13 | −2.07±.75 | −1.56±.57 |
| DESPF | −24.30±10.89 | −12.66±30.16 | −25.31±60.32 | −16.88±40.21 | −19.02±45.32 | −35.31±14.77 | −26.75±64.10 | −53.51±128.19 | −35.67±85.46 | −32.19±77.12 | −42.60±11.13 | −32.56±75.96 | −65.12±151.92 | −43.41±101.28 | −32.67±76.21 |
| PRIME | −58.62±3.36 | −5.85±.65 | −11.70±1.30 | −7.80±.87 | −8.79±.98 | −58.62±3.36 | −6.06±.67 | −12.12±1.34 | −8.08±.90 | −7.29±.81 | −58.62±3.36 | −6.13±.68 | −12.26±1.36 | −8.17±.91 | −6.15±.68 |

Table 68: Overall results for CEC constrained tasks with 128 solutions and 100th percentile evaluations. In this case, 0% of the values are missing near the worst value and another 20% near the optimal value. Details are the same as Table 5.

| Steps | t = 50 | | | | | t = 100 | | | | | t = 150 | | | | |
|---|---|---|---|---|---|---|---|---|---|---|---|---|---|---|---|
| **Task** $f(x^*_{OFF})$ | CEC 1 −2.08e4±424.97 | | | | | | | | | | | | | | |
| Metric | FS | SI | OI | SO | SO_w | FS | SI | OI | SO | SO_w | FS | SI | OI | SO | SO_w |
| CARCOO | - | - | - | - | - | - | - | - | - | - | - | - | - | - | - |
| DEPF | −2.08e4±424.97 | 1.00±.00 | 1.00±.00 | 1.00±.00 | 1.00±.00 | −2.08e4±424.97 | 1.00±.00 | 1.00±.00 | 1.00±.00 | 1.00±.00 | −2.08e4±424.97 | 1.00±.00 | 1.00±.00 | 1.00±.00 | 1.00±.00 |
| DESPF | −2.08e4±424.97 | 1.00±.00 | 1.00±.00 | 1.00±.00 | 1.00±.00 | −2.08e4±424.97 | 1.00±.00 | 1.00±.00 | 1.00±.00 | 1.00±.00 | −2.08e4±424.97 | 1.00±.00 | 1.00±.00 | 1.00±.00 | 1.00±.00 |
| PRIME | - | - | - | - | - | - | - | - | - | - | - | - | - | - | - |
| **Task** $f(x^*_{OFF})$ | CEC 2 −1.08±0.00 | | | | | | | | | | | | | | |
| CARCOO | −2.04±.00 | −93.25±.99 | −93.25±.99 | −93.25±.99 | −93.25±.99 | −2.04±.00 | −94.27±.72 | −94.27±.72 | −94.27±.72 | −94.27±.72 | −2.04±.00 | −94.60±.67 | −94.60±.67 | −94.60±.67 | −94.60±.67 |
| DEPF | −1.25±.06 | −20.27±3.74 | −20.27±3.74 | −20.27±3.74 | −20.27±3.74 | −1.36±.26 | −21.34±7.43 | −21.34±7.43 | −21.34±7.43 | −21.34±7.43 | −1.36±.26 | −23.28±13.72 | −23.28±13.72 | −23.28±13.72 | −23.28±13.72 |
| DESPF | −1.39±.25 | −26.59±10.96 | −26.59±10.96 | −26.59±10.96 | −26.59±10.96 | −1.47±.33 | −28.66±17.99 | −28.66±17.99 | −28.66±17.99 | −28.66±17.99 | −1.47±.33 | −29.89±20.56 | −29.89±20.56 | −29.89±20.56 | −29.89±20.56 |
| PRIME | −2.04±.00 | −82.31±5.36 | −82.31±5.36 | −82.31±5.36 | −82.31±5.36 | −2.04±.00 | −88.85±2.74 | −88.85±2.74 | −88.85±2.74 | −88.85±2.74 | −2.04±.00 | −91.00±1.90 | −91.00±1.90 | −91.00±1.90 | −91.00±1.90 |
| **Task** $f(x^*_{OFF})$ | CEC 3 −2.0±0.00 | | | | | | | | | | | | | | |
| CARCOO | −2.30±.72 | −16.57±41.38 | −16.52±41.40 | −16.55±41.39 | −16.54±41.39 | −2.30±.72 | −22.70±56.81 | −22.65±56.83 | −22.68±56.82 | −22.68±56.81 | −2.30±.72 | −24.72±61.88 | −24.68±61.90 | −24.70±61.89 | −24.72±61.88 |
| DEPF | −3.19±.04 | −74.85±13.36 | −76.48±14.04 | −75.60±13.51 | −75.88±13.04 | −3.31±.34 | −95.42±2.19 | −97.33±7.08 | −96.30±3.66 | −96.00±3.75 | −3.31±.34 | −106.36±15.52 | −108.37±14.61 | −107.29±14.87 | −106.37±15.51 |
| DESPF | −2.90±.44 | −57.97±31.82 | −57.97±31.82 | −57.97±31.82 | −57.97±31.82 | −3.31±.34 | −84.44±21.35 | −84.44±21.35 | −84.44±21.35 | −84.44±21.35 | −3.31±.34 | −99.67±21.76 | −99.67±21.76 | −99.67±21.76 | −99.67±21.76 |
| PRIME | −3.93±.70 | −198.38±24.28 | −199.89±22.34 | −199.11±23.28 | −199.37±22.98 | −4.05±.08 | −188.74±27.40 | −190.20±25.96 | −189.44±26.65 | −189.21±26.89 | −4.20±.00 | −192.25±18.70 | −193.85±17.48 | −193.03±18.00 | −192.26±18.69 |
| **Task** $f(x^*_{OFF})$ | CEC 4 −264.43±0.30 | | | | | | | | | | | | | | |
| CARCOO | −379.59±1.78 | −1.14e4±195.53 | −1.14e4±195.53 | −1.14e4±195.53 | −1.14e4±195.53 | −379.59±1.78 | −1.15e4±196.54 | −1.15e4±196.54 | −1.15e4±196.54 | −1.15e4±196.54 | −379.59±1.78 | −1.15e4±196.88 | −1.15e4±196.88 | −1.15e4±196.88 | −1.15e4±196.88 |
| DEPF | −298.17±16.88 | −1.82e3±740.14 | −1.84e3±708.61 | −1.83e3±723.23 | −1.83e3±723.25 | −310.99±16.02 | −2.61e3±978.91 | −2.63e3±919.91 | −2.61e3±957.85 | −2.61e3±969.29 | −330.73±23.38 | −3.44e3±1.37e3 | −3.46e3±1.32e3 | −3.45e3±1.35e3 | −3.44e3±1.37e3 |
| DESPF | −295.95±11.38 | −1.64e3±685.73 | −1.64e3±651.65 | −1.64e3±684.36 | −1.64e3±685.67 | −304.31±12.67 | −2.37e3±981.18 | −2.38e3±974.65 | −2.37e3±979.00 | −2.37e3±979.85 | −308.01±11.13 | −2.74e3±1.08e3 | −2.74e3±1.07e3 | −2.74e3±1.07e3 | −2.74e3±1.08e3 |
| PRIME | −303.25±44.32 | −7.63e3±4.18e3 | −7.86e3±3.89e3 | −7.71e3±4.08e3 | −7.74e3±4.03e3 | −349.60±42.97 | −6.88e3±3.49e3 | −7.12e3±3.76e3 | −6.96e3±3.55e3 | −6.93e3±3.59e3 | −367.20±32.62 | −6.55e3±3.90e3 | −6.85e3±3.62e3 | −6.65e3±3.76e3 | −6.55e3±3.90e3 |
| **Task** $f(x^*_{OFF})$ | CEC 5 −4.91±0.10 | | | | | | | | | | | | | | |
| CARCOO | −58.59±2.09 | −8.78±.09 | −17.56±1.97 | −11.71±1.31 | −13.19±1.48 | −58.59±2.09 | −9.12±1.00 | −18.24±2.00 | −12.16±1.33 | −10.97±1.20 | −58.59±2.09 | −9.23±1.00 | −18.46±2.00 | −12.31±1.34 | −9.26±1.01 |
| DEPF | −18.29±3.27 | −94.30±246.06 | −96.09±243.40 | −94.89±243.84 | −95.20±243.73 | −24.29±3.94 | −158.77±407.39 | −162.97±405.84 | −160.17±406.87 | −159.63±407.07 | −31.38±7.06 | −217.94±559.47 | −223.62±557.37 | −219.84±558.77 | −217.96±559.47 |
| DESPF | −24.53±2.58 | −411.88±534.56 | −414.74±532.40 | −412.83±533.84 | −413.31±533.48 | −25.96±4.06 | −657.51±871.76 | −661.39±868.86 | −658.80±870.79 | −658.30±871.17 | −31.88±8.50 | −777.29±1.02e3 | −781.59±1.02e3 | −778.73±1.02e3 | −777.31±1.02e3 |
| PRIME | −58.62±3.36 | −12.93±2.02 | −25.86±4.04 | −17.24±2.70 | −19.43±3.04 | −58.62±3.36 | −13.19±2.03 | −26.37±4.06 | −17.58±2.71 | −15.87±2.44 | −58.62±3.36 | −13.27±2.03 | −26.54±4.07 | −17.69±2.71 | −13.31±2.04 |

Table 69: Overall results for CEC constrained tasks with 128 solutions and 100th percentile evaluations. In this case, 0% of the values are missing near the worst value and another 10% near the optimal value. Details are the same as Table 5.

| Steps | t = 50 | | | | | t = 100 | | | | | t = 150 | | | | |
|---|---|---|---|---|---|---|---|---|---|---|---|---|---|---|---|
| **Metric** | FS | SI | OI | SO | $SO_\omega$ | FS | SI | OI | SO | $SO_\omega$ | FS | SI | OI | SO | $SO_\omega$ |
| **CEC 1** $f(x^*_{OFF})$ = $-2.08e4_{\pm424.97}$ | | | | | | | | | | | | | | | |
| CARCOO | - | - | - | - | - | - | - | - | - | - | - | - | - | - | - |
| DEPF | $-2.08e4_{\pm424.97}$ | $1.00_{\pm0.00}$ | $1.00_{\pm0.00}$ | $1.00_{\pm0.00}$ | $1.00_{\pm0.00}$ | $-2.08e4_{\pm424.97}$ | $1.00_{\pm0.00}$ | $1.00_{\pm0.00}$ | $1.00_{\pm0.00}$ | $1.00_{\pm0.00}$ | $-2.08e4_{\pm424.97}$ | $1.00_{\pm0.00}$ | $1.00_{\pm0.00}$ | $1.00_{\pm0.00}$ | $1.00_{\pm0.00}$ |
| DESPF | $-2.08e4_{\pm424.97}$ | $1.00_{\pm0.00}$ | $1.00_{\pm0.00}$ | $1.00_{\pm0.00}$ | $1.00_{\pm0.00}$ | $-2.08e4_{\pm424.97}$ | $1.00_{\pm0.00}$ | $1.00_{\pm0.00}$ | $1.00_{\pm0.00}$ | $1.00_{\pm0.00}$ | $-2.08e4_{\pm424.97}$ | $1.00_{\pm0.00}$ | $1.00_{\pm0.00}$ | $1.00_{\pm0.00}$ | $1.00_{\pm0.00}$ |
| PRIME | - | - | - | - | - | - | - | - | - | - | - | - | - | - | - |
| **CEC 2** $f(x^*_{OFF})$ = $-1.08_{\pm0.00}$ | | | | | | | | | | | | | | | |
| CARCOO | $-2.04_{\pm0.00}$ | $-93.67_{\pm1.70}$ | $-93.67_{\pm1.70}$ | $-93.67_{\pm1.70}$ | $-93.67_{\pm1.70}$ | $-2.04_{\pm0.00}$ | $-94.47_{\pm1.14}$ | $-94.47_{\pm1.14}$ | $-94.47_{\pm1.14}$ | $-94.47_{\pm1.14}$ | $-2.04_{\pm0.00}$ | $-94.74_{\pm0.96}$ | $-94.74_{\pm0.96}$ | $-94.74_{\pm0.96}$ | $-94.74_{\pm0.96}$ |
| DEPF | $-1.25_{\pm0.00}$ | $-17.92_{\pm1.80}$ | $-17.92_{\pm1.80}$ | $-17.92_{\pm1.80}$ | $-17.92_{\pm1.80}$ | $-1.35_{\pm0.26}$ | $-16.88_{\pm0.84}$ | $-16.88_{\pm0.84}$ | $-16.88_{\pm0.84}$ | $-16.88_{\pm0.84}$ | $-1.36_{\pm0.26}$ | $-20.06_{\pm8.82}$ | $-20.06_{\pm8.82}$ | $-20.06_{\pm8.82}$ | $-20.06_{\pm8.82}$ |
| DESPF | $-1.36_{\pm0.26}$ | $-26.33_{\pm16.84}$ | $-26.33_{\pm16.84}$ | $-26.33_{\pm16.84}$ | $-26.33_{\pm16.84}$ | $-1.36_{\pm0.26}$ | $-26.64_{\pm21.35}$ | $-26.64_{\pm21.35}$ | $-26.64_{\pm21.35}$ | $-26.64_{\pm21.35}$ | $-1.36_{\pm0.26}$ | $-26.74_{\pm22.86}$ | $-26.74_{\pm22.86}$ | $-26.74_{\pm22.86}$ | $-26.74_{\pm22.86}$ |
| PRIME | $-2.04_{\pm0.00}$ | $-84.99_{\pm4.41}$ | $-84.99_{\pm4.41}$ | $-84.99_{\pm4.41}$ | $-84.99_{\pm4.41}$ | $-2.04_{\pm0.00}$ | $-90.18_{\pm2.00}$ | $-90.18_{\pm2.00}$ | $-90.18_{\pm2.00}$ | $-90.18_{\pm2.00}$ | $-2.04_{\pm0.00}$ | $-91.89_{\pm1.23}$ | $-91.89_{\pm1.23}$ | $-91.89_{\pm1.23}$ | $-91.89_{\pm1.23}$ |
| **CEC 3** $f(x^*_{OFF})$ = $-2.0_{\pm0.00}$ | | | | | | | | | | | | | | | |
| CARCOO | $-2.06_{\pm0.03}$ | $-4.77_{\pm2.96}$ | $-4.75_{\pm2.98}$ | $-4.76_{\pm2.97}$ | $-4.76_{\pm2.98}$ | $-2.07_{\pm0.03}$ | $-5.08_{\pm3.11}$ | $-5.07_{\pm3.13}$ | $-5.08_{\pm3.12}$ | $-5.08_{\pm3.12}$ | $-2.07_{\pm0.03}$ | $-5.20_{\pm3.17}$ | $-5.19_{\pm3.19}$ | $-5.19_{\pm3.18}$ | $-5.20_{\pm3.17}$ |
| DEPF | $-3.03_{\pm0.61}$ | $-62.34_{\pm27.40}$ | $-62.64_{\pm27.34}$ | $-62.49_{\pm27.37}$ | $-62.54_{\pm27.36}$ | $-3.30_{\pm0.34}$ | $-91.27_{\pm32.28}$ | $-91.76_{\pm32.16}$ | $-91.51_{\pm32.21}$ | $-91.43_{\pm32.23}$ | $-3.30_{\pm0.34}$ | $-103.72_{\pm31.50}$ | $-104.27_{\pm31.32}$ | $-103.99_{\pm31.41}$ | $-103.72_{\pm31.49}$ |
| DESPF | $-2.73_{\pm0.53}$ | $-32.86_{\pm23.70}$ | $-32.88_{\pm23.70}$ | $-32.87_{\pm23.70}$ | $-32.87_{\pm23.70}$ | $-3.17_{\pm0.06}$ | $-68.38_{\pm15.85}$ | $-68.42_{\pm15.84}$ | $-68.40_{\pm15.84}$ | $-68.39_{\pm15.84}$ | $-3.04_{\pm0.34}$ | $-80.80_{\pm16.10}$ | $-80.83_{\pm16.09}$ | $-80.81_{\pm16.09}$ | $-80.80_{\pm16.10}$ |
| PRIME | $-3.41_{\pm1.01}$ | $-184.37_{\pm41.14}$ | $-184.37_{\pm41.14}$ | $-184.37_{\pm41.14}$ | $-184.37_{\pm41.14}$ | $-3.44_{\pm0.99}$ | $-173.44_{\pm35.90}$ | $-173.44_{\pm35.90}$ | $-173.44_{\pm35.90}$ | $-173.44_{\pm35.90}$ | $-4.20_{\pm0.00}$ | $-184.68_{\pm24.27}$ | $-184.68_{\pm24.27}$ | $-184.68_{\pm24.27}$ | $-184.68_{\pm24.27}$ |
| **CEC 4** $f(x^*_{OFF})$ = $-264.43_{\pm0.30}$ | | | | | | | | | | | | | | | |
| CARCOO | $-379.59_{\pm1.78}$ | $-1.14e4_{\pm195.53}$ | $-1.14e4_{\pm195.53}$ | $-1.14e4_{\pm195.53}$ | $-1.14e4_{\pm195.53}$ | $-379.59_{\pm1.78}$ | $-1.15e4_{\pm196.54}$ | $-1.15e4_{\pm196.54}$ | $-1.15e4_{\pm196.54}$ | $-1.15e4_{\pm196.54}$ | $-379.59_{\pm1.78}$ | $-1.15e4_{\pm196.88}$ | $-1.15e4_{\pm196.88}$ | $-1.15e4_{\pm196.88}$ | $-1.15e4_{\pm196.88}$ |
| DEPF | $-302.23_{\pm9.88}$ | $-1.58e3_{\pm905.98}$ | $-1.59e3_{\pm802.37}$ | $-1.58e3_{\pm801.43}$ | $-1.58e3_{\pm890.13}$ | $-323.60_{\pm20.38}$ | $-2.66e3_{\pm1.14e3}$ | $-2.68e3_{\pm1.50e3}$ | $-2.67e3_{\pm1.53e3}$ | $-2.66e3_{\pm1.53e3}$ | $-340.48_{\pm13.95}$ | $-3.36e3_{\pm1.99e3}$ | $-3.40e3_{\pm1.94e3}$ | $-3.37e3_{\pm1.97e3}$ | $-3.36e3_{\pm1.99e3}$ |
| DESPF | $-296.77_{\pm11.49}$ | $-2.02e3_{\pm447.63}$ | $-2.02e3_{\pm447.63}$ | $-2.02e3_{\pm447.63}$ | $-2.02e3_{\pm447.63}$ | $-308.98_{\pm17.20}$ | $-2.95e3_{\pm807.57}$ | $-2.95e3_{\pm807.57}$ | $-2.95e3_{\pm807.57}$ | $-2.95e3_{\pm807.57}$ | $-326.85_{\pm24.93}$ | $-3.73e3_{\pm1.01e3}$ | $-3.73e3_{\pm1.01e3}$ | $-3.73e3_{\pm1.01e3}$ | $-3.73e3_{\pm1.01e3}$ |
| PRIME | $-340.69_{\pm49.51}$ | $-9.55e3_{\pm3.63e3}$ | $-9.56e3_{\pm3.60e3}$ | $-9.55e3_{\pm3.62e3}$ | $-9.55e3_{\pm3.62e3}$ | $-348.26_{\pm42.19}$ | $-8.81e3_{\pm3.84e3}$ | $-8.82e3_{\pm3.81e3}$ | $-8.81e3_{\pm3.83e3}$ | $-8.81e3_{\pm3.84e3}$ | $-379.59_{\pm1.78}$ | $-8.70e3_{\pm3.91e3}$ | $-8.72e3_{\pm3.88e3}$ | $-8.71e3_{\pm3.90e3}$ | $-8.70e3_{\pm3.91e3}$ |
| **CEC 5** $f(x^*_{OFF})$ = $-2.95_{\pm0.06}$ | | | | | | | | | | | | | | | |
| CARCOO | $-58.62_{\pm3.36}$ | $-42.35_{\pm14.52}$ | $-84.71_{\pm29.04}$ | $-56.47_{\pm19.36}$ | $-63.64_{\pm21.81}$ | $-58.62_{\pm3.36}$ | $-43.42_{\pm14.95}$ | $-86.83_{\pm29.80}$ | $-57.89_{\pm19.93}$ | $-52.24_{\pm17.98}$ | $-58.62_{\pm3.36}$ | $-43.77_{\pm15.09}$ | $-87.53_{\pm30.17}$ | $-58.36_{\pm20.11}$ | $-43.91_{\pm15.14}$ |
| DEPF | $-18.29_{\pm4.16}$ | $-618.94_{\pm267.81}$ | $-624.02_{\pm256.95}$ | $-620.63_{\pm264.16}$ | $-621.49_{\pm262.32}$ | $-28.55_{\pm6.75}$ | $-1.21e3_{\pm500.57}$ | $-1.22e3_{\pm478.68}$ | $-1.21e3_{\pm493.23}$ | $-1.21e3_{\pm496.09}$ | $-29.27_{\pm7.27}$ | $-1.56e3_{\pm689.54}$ | $-1.57e3_{\pm634.27}$ | $-1.57e3_{\pm651.07}$ | $-1.56e3_{\pm659.45}$ |
| DESPF | $-21.82_{\pm4.83}$ | $-1.28e3_{\pm154.91}$ | $-1.28e3_{\pm154.91}$ | $-1.28e3_{\pm154.91}$ | $-1.28e3_{\pm154.91}$ | $-30.30_{\pm7.29}$ | $-1.77e3_{\pm346.09}$ | $-1.77e3_{\pm346.09}$ | $-1.77e3_{\pm346.09}$ | $-1.77e3_{\pm346.09}$ | $-33.42_{\pm6.99}$ | $-2.13e3_{\pm455.78}$ | $-2.13e3_{\pm455.78}$ | $-2.13e3_{\pm455.78}$ | $-2.13e3_{\pm455.78}$ |
| PRIME | $-58.62_{\pm3.36}$ | $-4.82e3_{\pm1.84e3}$ | $-4.83e3_{\pm1.83e3}$ | $-4.82e3_{\pm1.84e3}$ | $-4.83e3_{\pm1.84e3}$ | $-58.62_{\pm3.36}$ | $-4.85e3_{\pm1.85e3}$ | $-4.85e3_{\pm1.84e3}$ | $-4.85e3_{\pm1.85e3}$ | $-4.85e3_{\pm1.85e3}$ | $-58.62_{\pm3.36}$ | $-4.86e3_{\pm1.86e3}$ | $-4.86e3_{\pm1.85e3}$ | $-4.86e3_{\pm1.85e3}$ | $-4.86e3_{\pm1.86e3}$ |

Table 70: Overall results for CEC constrained tasks with 128 solutions and 100th percentile evaluations. In this case, 10% of the values are missing near the worst value and another 40% near the optimal value. Details are the same as Table 5.

| Steps | t = 50 | | | | | t = 100 | | | | | t = 150 | | | | |
|---|---|---|---|---|---|---|---|---|---|---|---|---|---|---|---|
| **Metric** | FS | SI | OI | SO | $SO_\omega$ | FS | SI | OI | SO | $SO_\omega$ | FS | SI | OI | SO | $SO_\omega$ |
| **CEC 1** $f(x^*_{OFF})$ = $-1.39e4_{\pm94.68}$ | | | | | | | | | | | | | | | |
| CARCOO | - | - | - | - | - | - | - | - | - | - | - | - | - | - | - |
| DEPF | $-1.39e4_{\pm94.68}$ | $1.00_{\pm0.00}$ | $1.00_{\pm0.00}$ | $1.00_{\pm0.00}$ | $1.00_{\pm0.00}$ | $-1.39e4_{\pm94.68}$ | $1.00_{\pm0.00}$ | $1.00_{\pm0.00}$ | $1.00_{\pm0.00}$ | $1.00_{\pm0.00}$ | $-1.39e4_{\pm94.68}$ | $1.00_{\pm0.00}$ | $1.00_{\pm0.00}$ | $1.00_{\pm0.00}$ | $1.00_{\pm0.00}$ |
| DESPF | $-1.39e4_{\pm94.68}$ | $1.00_{\pm0.00}$ | $1.00_{\pm0.00}$ | $1.00_{\pm0.00}$ | $1.00_{\pm0.00}$ | $-1.39e4_{\pm94.68}$ | $1.00_{\pm0.00}$ | $1.00_{\pm0.00}$ | $1.00_{\pm0.00}$ | $1.00_{\pm0.00}$ | $-1.39e4_{\pm94.68}$ | $1.00_{\pm0.00}$ | $1.00_{\pm0.00}$ | $1.00_{\pm0.00}$ | $1.00_{\pm0.00}$ |
| PRIME | - | - | - | - | - | - | - | - | - | - | - | - | - | - | - |
| **CEC 2** $f(x^*_{OFF})$ = $-1.08_{\pm0.00}$ | | | | | | | | | | | | | | | |
| CARCOO | $-1.23_{\pm0.10}$ | $-12.31_{\pm9.89}$ | $-12.31_{\pm9.89}$ | $-12.31_{\pm9.89}$ | $-12.31_{\pm9.89}$ | $-1.25_{\pm0.11}$ | $-13.57_{\pm9.82}$ | $-13.57_{\pm9.82}$ | $-13.57_{\pm9.82}$ | $-13.57_{\pm9.82}$ | $-1.25_{\pm0.11}$ | $-14.18_{\pm9.98}$ | $-14.18_{\pm9.98}$ | $-14.18_{\pm9.98}$ | $-14.18_{\pm9.98}$ |
| DEPF | $-1.26_{\pm0.03}$ | $-15.68_{\pm0.82}$ | $-15.68_{\pm0.82}$ | $-15.68_{\pm0.82}$ | $-15.68_{\pm0.82}$ | $-1.27_{\pm0.04}$ | $-16.39_{\pm1.47}$ | $-16.39_{\pm1.47}$ | $-16.39_{\pm1.47}$ | $-16.39_{\pm1.47}$ | $-1.28_{\pm0.04}$ | $-17.23_{\pm2.06}$ | $-17.23_{\pm2.06}$ | $-17.23_{\pm2.06}$ | $-17.23_{\pm2.06}$ |
| DESPF | $-1.28_{\pm0.04}$ | $-17.89_{\pm2.62}$ | $-17.89_{\pm2.62}$ | $-17.89_{\pm2.62}$ | $-17.89_{\pm2.62}$ | $-1.28_{\pm0.04}$ | $-18.48_{\pm3.25}$ | $-18.48_{\pm3.25}$ | $-18.48_{\pm3.25}$ | $-18.48_{\pm3.25}$ | $-1.28_{\pm0.04}$ | $-18.68_{\pm3.52}$ | $-18.68_{\pm3.52}$ | $-18.68_{\pm3.52}$ | $-18.68_{\pm3.52}$ |
| PRIME | $-1.34_{\pm0.00}$ | $-22.77_{\pm2.06}$ | $-23.50_{\pm2.67}$ | $-23.08_{\pm1.27}$ | $-23.21_{\pm0.97}$ | $-1.34_{\pm0.00}$ | $-23.27_{\pm2.04}$ | $-24.02_{\pm0.38}$ | $-23.59_{\pm1.22}$ | $-23.48_{\pm1.51}$ | $-1.34_{\pm0.00}$ | $-23.44_{\pm2.04}$ | $-24.19_{\pm0.36}$ | $-23.76_{\pm1.21}$ | $-23.44_{\pm2.03}$ |
| **CEC 3** $f(x^*_{OFF})$ = $-2.12_{\pm0.02}$ | | | | | | | | | | | | | | | |
| CARCOO | $-3.36_{\pm0.05}$ | $-19.50_{\pm38.72}$ | $-23.75_{\pm37.12}$ | $-20.91_{\pm38.19}$ | $-21.63_{\pm37.92}$ | $-3.36_{\pm0.05}$ | $-20.00_{\pm39.69}$ | $-24.38_{\pm38.08}$ | $-21.46_{\pm39.14}$ | $-20.89_{\pm39.35}$ | $-3.36_{\pm0.05}$ | $-20.17_{\pm40.00}$ | $-24.59_{\pm38.35}$ | $-21.64_{\pm39.48}$ | $-20.18_{\pm40.00}$ |
| DEPF | $-3.20_{\pm0.00}$ | $-2.55_{\pm0.40}$ | $-5.11_{\pm1.62}$ | $-3.40_{\pm1.08}$ | $-3.84_{\pm1.21}$ | $-3.20_{\pm0.00}$ | $-3.64_{\pm0.74}$ | $-7.28_{\pm1.49}$ | $-4.85_{\pm0.99}$ | $-4.38_{\pm0.89}$ | $-3.20_{\pm0.00}$ | $-3.99_{\pm0.78}$ | $-7.99_{\pm1.56}$ | $-5.33_{\pm1.04}$ | $-4.01_{\pm0.79}$ |
| DESPF | $-3.06_{\pm0.33}$ | $-9.00_{\pm16.91}$ | $-13.19_{\pm21.18}$ | $-10.59_{\pm18.84}$ | $-11.31_{\pm19.59}$ | $-3.20_{\pm0.00}$ | $-11.75_{\pm20.73}$ | $-17.48_{\pm25.69}$ | $-13.90_{\pm23.02}$ | $-13.10_{\pm22.23}$ | $-3.20_{\pm0.00}$ | $-12.85_{\pm22.54}$ | $-19.18_{\pm27.88}$ | $-15.23_{\pm25.02}$ | $-12.88_{\pm22.57}$ |
| PRIME | $-2.88_{\pm0.46}$ | $-2.74_{\pm0.78}$ | $-5.48_{\pm1.55}$ | $-3.65_{\pm1.04}$ | $-4.11_{\pm1.17}$ | $-3.36_{\pm0.05}$ | $-3.55_{\pm0.51}$ | $-7.09_{\pm1.03}$ | $-4.73_{\pm0.68}$ | $-4.27_{\pm0.62}$ | $-3.36_{\pm0.05}$ | $-3.97_{\pm0.52}$ | $-7.94_{\pm1.04}$ | $-5.29_{\pm0.69}$ | $-3.98_{\pm0.52}$ |
| **CEC 4** $f(x^*_{OFF})$ = $-264.43_{\pm0.30}$ | | | | | | | | | | | | | | | |
| CARCOO | $-327.69_{\pm1.16}$ | $-6.26e3_{\pm110.35}$ | $-6.26e3_{\pm110.35}$ | $-6.26e3_{\pm110.35}$ | $-6.26e3_{\pm110.35}$ | $-327.69_{\pm1.16}$ | $-6.29e3_{\pm110.92}$ | $-6.29e3_{\pm110.92}$ | $-6.29e3_{\pm110.92}$ | $-6.29e3_{\pm110.92}$ | $-327.69_{\pm1.16}$ | $-6.30e3_{\pm111.11}$ | $-6.30e3_{\pm111.11}$ | $-6.30e3_{\pm111.11}$ | $-6.30e3_{\pm111.11}$ |
| DEPF | $-297.09_{\pm11.25}$ | $-1.97e3_{\pm611.15}$ | $-1.97e3_{\pm611.15}$ | $-1.97e3_{\pm611.15}$ | $-1.97e3_{\pm611.15}$ | $-309.86_{\pm15.06}$ | $-2.95e3_{\pm887.97}$ | $-2.95e3_{\pm887.97}$ | $-2.95e3_{\pm887.97}$ | $-2.95e3_{\pm887.97}$ | $-323.94_{\pm12.51}$ | $-3.64e3_{\pm900.02}$ | $-3.64e3_{\pm900.02}$ | $-3.64e3_{\pm900.02}$ | $-3.64e3_{\pm900.02}$ |
| DESPF | $-292.79_{\pm11.36}$ | $-1.60e3_{\pm688.42}$ | $-1.63e3_{\pm638.42}$ | $-1.61e3_{\pm671.42}$ | $-1.61e3_{\pm662.92}$ | $-307.21_{\pm18.96}$ | $-2.47e3_{\pm1.07e3}$ | $-2.52e3_{\pm984.76}$ | $-2.49e3_{\pm1.04e3}$ | $-2.48e3_{\pm1.05e3}$ | $-315.90_{\pm19.16}$ | $-3.04e3_{\pm1.45e3}$ | $-3.08e3_{\pm1.31e3}$ | $-3.05e3_{\pm1.42e3}$ | $-3.04e3_{\pm1.45e3}$ |
| PRIME | $-327.69_{\pm1.16}$ | $-6.26e3_{\pm110.35}$ | $-6.26e3_{\pm110.35}$ | $-6.26e3_{\pm110.35}$ | $-6.26e3_{\pm110.35}$ | $-327.69_{\pm1.16}$ | $-6.25e3_{\pm91.62}$ | $-6.25e3_{\pm91.62}$ | $-6.25e3_{\pm91.62}$ | $-6.25e3_{\pm91.62}$ | $-327.69_{\pm1.16}$ | $-6.26e3_{\pm108.03}$ | $-6.26e3_{\pm108.03}$ | $-6.26e3_{\pm108.03}$ | $-6.26e3_{\pm108.03}$ |
| **CEC 5** $f(x^*_{OFF})$ = $-8.61_{\pm0.14}$ | | | | | | | | | | | | | | | |
| CARCOO | $-27.28_{\pm0.15}$ | $-0.80_{\pm0.07}$ | $-1.59_{\pm0.13}$ | $-1.06_{\pm0.09}$ | $-1.20_{\pm0.10}$ | $-27.28_{\pm0.15}$ | $-0.91_{\pm0.06}$ | $-1.82_{\pm0.11}$ | $-1.21_{\pm0.07}$ | $-1.09_{\pm0.07}$ | $-27.28_{\pm0.15}$ | $-0.94_{\pm0.08}$ | $-1.89_{\pm0.11}$ | $-1.26_{\pm0.07}$ | $-0.95_{\pm0.08}$ |
| DEPF | $-15.85_{\pm5.31}$ | $0.39_{\pm0.15}$ | $0.78_{\pm0.29}$ | $0.52_{\pm0.20}$ | $0.58_{\pm0.22}$ | $-17.88_{\pm4.64}$ | $-0.04_{\pm0.30}$ | $-0.08_{\pm0.61}$ | $-0.05_{\pm0.41}$ | $-0.05_{\pm0.47}$ | $-23.33_{\pm3.35}$ | $-0.32_{\pm0.35}$ | $-0.63_{\pm0.70}$ | $-0.42_{\pm0.47}$ | $-0.32_{\pm0.53}$ |
| DESPF | $-25.22_{\pm3.64}$ | $-0.91_{\pm0.40}$ | $-1.83_{\pm1.20}$ | $-1.22_{\pm0.80}$ | $-1.37_{\pm0.90}$ | $-27.27_{\pm0.14}$ | $-1.48_{\pm0.68}$ | $-2.97_{\pm1.35}$ | $-1.98_{\pm0.90}$ | $-1.78_{\pm0.81}$ | $-27.27_{\pm0.14}$ | $-1.68_{\pm0.71}$ | $-3.35_{\pm1.43}$ | $-2.23_{\pm0.95}$ | $-1.68_{\pm0.72}$ |
| PRIME | $-27.27_{\pm0.14}$ | $-0.98_{\pm0.14}$ | $-1.95_{\pm0.28}$ | $-1.30_{\pm0.18}$ | $-1.47_{\pm0.21}$ | $-27.27_{\pm0.14}$ | $-1.10_{\pm0.14}$ | $-2.20_{\pm0.27}$ | $-1.46_{\pm0.18}$ | $-1.32_{\pm0.16}$ | $-27.27_{\pm0.14}$ | $-1.14_{\pm0.14}$ | $-2.28_{\pm0.27}$ | $-1.52_{\pm0.18}$ | $-1.14_{\pm0.14}$ |

Table 71: Overall results for CEC constrained tasks with 128 solutions and 100th percentile evaluations. In this case, 20% of the values are missing near the worst value and another 30% near the optimal value. Details are the same as Table 5.

| Steps | t = 50 | | | | | t = 100 | | | | | t = 150 | | | | |
|---|---|---|---|---|---|---|---|---|---|---|---|---|---|---|---|
| **Metric** | FS | SI | OI | SO | $SO_\omega$ | FS | SI | OI | SO | $SO_\omega$ | FS | SI | OI | SO | $SO_\omega$ |
| **CEC 1** $f(x^*_{OFF})$ = $-1.17e4_{\pm74.77}$ | | | | | | | | | | | | | | | |
| CARCOO | - | - | - | - | - | - | - | - | - | - | - | - | - | - | - |
| DEPF | $-1.17e4_{\pm74.77}$ | $1.00_{\pm0.00}$ | $1.00_{\pm0.00}$ | $1.00_{\pm0.00}$ | $1.00_{\pm0.00}$ | $-1.17e4_{\pm74.77}$ | $1.00_{\pm0.00}$ | $1.00_{\pm0.00}$ | $1.00_{\pm0.00}$ | $1.00_{\pm0.00}$ | $-1.17e4_{\pm74.77}$ | $1.00_{\pm0.00}$ | $1.00_{\pm0.00}$ | $1.00_{\pm0.00}$ | $1.00_{\pm0.00}$ |
| DESPF | $-1.17e4_{\pm74.77}$ | $1.00_{\pm0.00}$ | $1.00_{\pm0.00}$ | $1.00_{\pm0.00}$ | $1.00_{\pm0.00}$ | $-1.17e4_{\pm74.77}$ | $1.00_{\pm0.00}$ | $1.00_{\pm0.00}$ | $1.00_{\pm0.00}$ | $1.00_{\pm0.00}$ | $-1.17e4_{\pm74.77}$ | $1.00_{\pm0.00}$ | $1.00_{\pm0.00}$ | $1.00_{\pm0.00}$ | $1.00_{\pm0.00}$ |
| PRIME | - | - | - | - | - | - | - | - | - | - | - | - | - | - | - |
| **CEC 2** $f(x^*_{OFF})$ = $-1.08_{\pm0.00}$ | | | | | | | | | | | | | | | |
| CARCOO | $-1.29_{\pm0.03}$ | $-19.66_{\pm2.42}$ | $-19.66_{\pm2.42}$ | $-19.66_{\pm2.42}$ | $-19.66_{\pm2.42}$ | $-1.30_{\pm0.00}$ | $-20.24_{\pm1.42}$ | $-20.24_{\pm1.42}$ | $-20.24_{\pm1.42}$ | $-20.24_{\pm1.42}$ | $-1.30_{\pm0.00}$ | $-20.49_{\pm0.95}$ | $-20.49_{\pm0.95}$ | $-20.49_{\pm0.95}$ | $-20.49_{\pm0.95}$ |
| DEPF | $-1.25_{\pm0.00}$ | $-15.64_{\pm0.55}$ | $-15.64_{\pm0.55}$ | $-15.64_{\pm0.55}$ | $-15.64_{\pm0.55}$ | $-1.26_{\pm0.02}$ | $-15.99_{\pm0.76}$ | $-15.99_{\pm0.76}$ | $-15.99_{\pm0.76}$ | $-15.99_{\pm0.76}$ | $-1.27_{\pm0.01}$ | $-16.55_{\pm1.15}$ | $-16.55_{\pm1.15}$ | $-16.55_{\pm1.15}$ | $-16.55_{\pm1.15}$ |
| DESPF | $-1.26_{\pm0.02}$ | $-16.97_{\pm1.99}$ | $-16.97_{\pm1.99}$ | $-16.97_{\pm1.99}$ | $-16.97_{\pm1.99}$ | $-1.27_{\pm0.03}$ | $-17.38_{\pm2.12}$ | $-17.38_{\pm2.12}$ | $-17.38_{\pm2.12}$ | $-17.38_{\pm2.12}$ | $-1.27_{\pm0.01}$ | $-17.52_{\pm2.20}$ | $-17.52_{\pm2.20}$ | $-17.52_{\pm2.20}$ | $-17.52_{\pm2.20}$ |
| PRIME | $-1.30_{\pm0.00}$ | $-20.43_{\pm0.37}$ | $-20.43_{\pm0.37}$ | $-20.43_{\pm0.37}$ | $-20.43_{\pm0.37}$ | $-1.30_{\pm0.00}$ | $-20.69_{\pm0.28}$ | $-20.69_{\pm0.28}$ | $-20.69_{\pm0.28}$ | $-20.69_{\pm0.28}$ | $-1.30_{\pm0.00}$ | $-20.78_{\pm0.25}$ | $-20.78_{\pm0.25}$ | $-20.78_{\pm0.25}$ | $-20.78_{\pm0.25}$ |
| **CEC 3** $f(x^*_{OFF})$ = $-2.0_{\pm0.00}$ | | | | | | | | | | | | | | | |
| CARCOO | $-2.90_{\pm0.33}$ | $-70.80_{\pm27.79}$ | $-75.11_{\pm29.02}$ | $-72.83_{\pm28.28}$ | $-73.57_{\pm28.50}$ | $-2.90_{\pm0.33}$ | $-77.10_{\pm29.66}$ | $-81.81_{\pm31.03}$ | $-79.32_{\pm30.22}$ | $-78.58_{\pm30.01}$ | $-2.90_{\pm0.33}$ | $-79.20_{\pm30.27}$ | $-84.05_{\pm31.68}$ | $-81.49_{\pm30.84}$ | $-79.23_{\pm30.28}$ |
| DEPF | $-2.78_{\pm0.41}$ | $-39.62_{\pm22.42}$ | $-39.87_{\pm22.50}$ | $-39.74_{\pm22.46}$ | $-39.79_{\pm22.47}$ | $-3.02_{\pm0.00}$ | $-65.06_{\pm20.79}$ | $-65.45_{\pm20.95}$ | $-65.25_{\pm20.87}$ | $-65.19_{\pm20.84}$ | $-3.02_{\pm0.00}$ | $-76.88_{\pm13.67}$ | $-77.31_{\pm13.84}$ | $-77.09_{\pm13.74}$ | $-76.88_{\pm13.67}$ |
| DESPF | $-2.77_{\pm0.42}$ | $-39.74_{\pm31.54}$ | $-41.30_{\pm32.77}$ | $-40.45_{\pm32.04}$ | $-40.72_{\pm32.26}$ | $-2.90_{\pm0.31}$ | $-62.23_{\pm26.47}$ | $-64.10_{\pm27.61}$ | $-63.08_{\pm26.88}$ | $-62.78_{\pm26.71}$ | $-3.02_{\pm0.00}$ | $-73.14_{\pm20.11}$ | $-75.12_{\pm21.06}$ | $-74.04_{\pm20.38}$ | $-73.15_{\pm20.11}$ |
| PRIME | $-2.71_{\pm0.41}$ | $-85.19_{\pm13.10}$ | $-85.19_{\pm13.10}$ | $-85.19_{\pm13.10}$ | $-85.19_{\pm13.10}$ | $-2.91_{\pm0.25}$ | $-88.42_{\pm8.35}$ | $-88.42_{\pm8.35}$ | $-88.42_{\pm8.35}$ | $-88.42_{\pm8.35}$ | $-3.02_{\pm0.01}$ | $-92.27_{\pm5.68}$ | $-92.27_{\pm5.68}$ | $-92.27_{\pm5.68}$ | $-92.27_{\pm5.68}$ |
| **CEC 4** $f(x^*_{OFF})$ = $-264.43_{\pm0.30}$ | | | | | | | | | | | | | | | |
| CARCOO | $-304.56_{\pm0.74}$ | $-3.97e3_{\pm82.73}$ | $-3.97e3_{\pm82.73}$ | $-3.97e3_{\pm82.73}$ | $-3.97e3_{\pm82.73}$ | $-304.56_{\pm0.74}$ | $-3.99e3_{\pm83.16}$ | $-3.99e3_{\pm83.16}$ | $-3.99e3_{\pm83.16}$ | $-3.99e3_{\pm83.16}$ | $-304.56_{\pm0.74}$ | $-4.00e3_{\pm83.30}$ | $-4.00e3_{\pm83.30}$ | $-4.00e3_{\pm83.30}$ | $-4.00e3_{\pm83.30}$ |
| DEPF | $-301.58_{\pm2.49}$ | $-2.31e3_{\pm396.37}$ | $-2.31e3_{\pm396.37}$ | $-2.31e3_{\pm396.37}$ | $-2.31e3_{\pm396.37}$ | $-305.77_{\pm12.31}$ | $-3.05e3_{\pm489.30}$ | $-3.05e3_{\pm489.30}$ | $-3.05e3_{\pm489.30}$ | $-3.05e3_{\pm489.30}$ | $-325.94_{\pm12.82}$ | $-3.77e3_{\pm574.70}$ | $-3.77e3_{\pm574.70}$ | $-3.77e3_{\pm574.70}$ | $-3.77e3_{\pm574.70}$ |
| DESPF | $-290.73_{\pm11.01}$ | $-1.78e3_{\pm478.30}$ | $-1.87e3_{\pm503.51}$ | $-1.81e3_{\pm414.88}$ | $-1.83e3_{\pm384.02}$ | $-302.77_{\pm7.45}$ | $-2.49e3_{\pm696.61}$ | $-2.61e3_{\pm469.78}$ | $-2.53e3_{\pm611.86}$ | $-2.51e3_{\pm443.97}$ | $-316.38_{\pm16.18}$ | $-3.04e3_{\pm670.73}$ | $-3.21e3_{\pm620.20}$ | $-3.09e3_{\pm768.74}$ | $-3.04e3_{\pm869.64}$ |
| PRIME | $-304.56_{\pm0.74}$ | $-3.97e3_{\pm82.73}$ | $-3.97e3_{\pm82.73}$ | $-3.97e3_{\pm82.73}$ | $-3.97e3_{\pm82.73}$ | $-304.56_{\pm0.74}$ | $-3.97e3_{\pm80.14}$ | $-3.97e3_{\pm80.14}$ | $-3.97e3_{\pm80.14}$ | $-3.97e3_{\pm80.14}$ | $-304.56_{\pm0.74}$ | $-3.99e3_{\pm79.31}$ | $-3.99e3_{\pm79.31}$ | $-3.99e3_{\pm79.31}$ | $-3.99e3_{\pm79.31}$ |
| **CEC 5** $f(x^*_{OFF})$ = $-6.75_{\pm0.10}$ | | | | | | | | | | | | | | | |
| CARCOO | - | - | - | - | - | - | - | - | - | - | - | - | - | - | - |
| DEPF | $-13.33_{\pm4.37}$ | $0.05_{\pm0.20}$ | $0.09_{\pm0.39}$ | $0.06_{\pm0.26}$ | $0.07_{\pm0.30}$ | $-19.28_{\pm1.48}$ | $-0.68_{\pm0.28}$ | $-1.36_{\pm0.56}$ | $-0.90_{\pm0.38}$ | $-0.82_{\pm0.34}$ | $-20.31_{\pm0.22}$ | $-1.08_{\pm0.34}$ | $-2.15_{\pm0.69}$ | $-1.44_{\pm0.46}$ | $-1.08_{\pm0.35}$ |
| DESPF | $-19.59_{\pm1.35}$ | $-3.95_{\pm5.68}$ | $-7.89_{\pm11.30}$ | $-5.26_{\pm7.53}$ | $-5.93_{\pm8.49}$ | $-20.13_{\pm0.57}$ | $-4.96_{\pm6.93}$ | $-9.93_{\pm13.86}$ | $-6.62_{\pm9.24}$ | $-5.97_{\pm8.34}$ | $-20.31_{\pm0.22}$ | $-5.31_{\pm7.34}$ | $-10.62_{\pm14.69}$ | $-7.08_{\pm9.79}$ | $-5.33_{\pm7.37}$ |
| PRIME | $-20.31_{\pm0.22}$ | $-1.07_{\pm0.15}$ | $-2.14_{\pm0.29}$ | $-1.43_{\pm0.20}$ | $-1.61_{\pm0.22}$ | $-20.31_{\pm0.22}$ | $-1.17_{\pm0.15}$ | $-2.34_{\pm0.30}$ | $-1.56_{\pm0.20}$ | $-1.41_{\pm0.18}$ | $-20.31_{\pm0.22}$ | $-1.21_{\pm0.15}$ | $-2.41_{\pm0.30}$ | $-1.61_{\pm0.20}$ | $-1.21_{\pm0.15}$ |

Table 72: Overall results for CEC constrained tasks with 128 solutions and 100th percentile evaluations. In this case, 30% of the values are missing near the worst value and another 20% near the optimal value. Details are the same as Table 5.

| Steps | t = 50 | | | | | t = 100 | | | | | t = 150 | | | | |
|---|---|---|---|---|---|---|---|---|---|---|---|---|---|---|---|
| **Task** $f(x^*_{OFF})$ | CEC 1 $-9.91e3_{\pm100.88}$ | | | | | | | | | | | | | | |
| Metric | FS | SI | OI | SO | SO$_\omega$ | FS | SI | OI | SO | SO$_\omega$ | FS | SI | OI | SO | SO$_\omega$ |
| CARCOO | - | - | - | - | - | - | - | - | - | - | - | - | - | - | - |
| DEPF | $-9.91e3_{\pm100.88}$ | $1.00_{\pm0.00}$ | $1.00_{\pm0.00}$ | $1.00_{\pm0.00}$ | $1.00_{\pm0.00}$ | $-9.91e3_{\pm100.88}$ | $1.00_{\pm0.00}$ | $1.00_{\pm0.00}$ | $1.00_{\pm0.00}$ | $1.00_{\pm0.00}$ | $-9.91e3_{\pm100.88}$ | $1.00_{\pm0.00}$ | $1.00_{\pm0.00}$ | $1.00_{\pm0.00}$ | $1.00_{\pm0.00}$ |
| DESPF | $-9.91e3_{\pm100.88}$ | $1.00_{\pm0.00}$ | $1.00_{\pm0.00}$ | $1.00_{\pm0.00}$ | $1.00_{\pm0.00}$ | $-9.91e3_{\pm100.88}$ | $1.00_{\pm0.00}$ | $1.00_{\pm0.00}$ | $1.00_{\pm0.00}$ | $1.00_{\pm0.00}$ | $-9.91e3_{\pm100.88}$ | $1.00_{\pm0.00}$ | $1.00_{\pm0.00}$ | $1.00_{\pm0.00}$ | $1.00_{\pm0.00}$ |
| PRIME | - | - | - | - | - | - | - | - | - | - | - | - | - | - | - |
| **Task** $f(x^*_{OFF})$ | CEC 2 $-1.08_{\pm0.00}$ | | | | | | | | | | | | | | |
| CARCOO | $-1.26_{\pm0.00}$ | $-16.37_{\pm0.27}$ | $-16.37_{\pm0.27}$ | $-16.37_{\pm0.27}$ | $-16.37_{\pm0.27}$ | $-1.26_{\pm0.00}$ | $-16.50_{\pm0.25}$ | $-16.50_{\pm0.25}$ | $-16.50_{\pm0.25}$ | $-16.50_{\pm0.25}$ | $-1.26_{\pm0.00}$ | $-16.54_{\pm0.24}$ | $-16.54_{\pm0.24}$ | $-16.54_{\pm0.24}$ | $-16.54_{\pm0.24}$ |
| DEPF | $-1.25_{\pm0.00}$ | $-15.33_{\pm0.48}$ | $-15.33_{\pm0.48}$ | $-15.33_{\pm0.48}$ | $-15.33_{\pm0.48}$ | $-1.25_{\pm0.01}$ | $-15.63_{\pm0.27}$ | $-15.63_{\pm0.27}$ | $-15.63_{\pm0.27}$ | $-15.63_{\pm0.27}$ | $-1.26_{\pm0.00}$ | $-15.85_{\pm0.17}$ | $-15.85_{\pm0.17}$ | $-15.85_{\pm0.17}$ | $-15.85_{\pm0.17}$ |
| DESPF | $-1.26_{\pm0.00}$ | $-16.29_{\pm0.29}$ | $-16.29_{\pm0.29}$ | $-16.29_{\pm0.29}$ | $-16.29_{\pm0.29}$ | $-1.26_{\pm0.00}$ | $-16.46_{\pm0.26}$ | $-16.46_{\pm0.26}$ | $-16.46_{\pm0.26}$ | $-16.46_{\pm0.26}$ | $-1.26_{\pm0.00}$ | $-16.51_{\pm0.25}$ | $-16.51_{\pm0.25}$ | $-16.51_{\pm0.25}$ | $-16.51_{\pm0.25}$ |
| PRIME | $-1.26_{\pm0.00}$ | $-15.51_{\pm0.69}$ | $-15.51_{\pm0.69}$ | $-15.51_{\pm0.69}$ | $-15.51_{\pm0.69}$ | $-1.26_{\pm0.00}$ | $-16.06_{\pm0.36}$ | $-16.06_{\pm0.36}$ | $-16.06_{\pm0.36}$ | $-16.06_{\pm0.36}$ | $-1.26_{\pm0.00}$ | $-16.25_{\pm0.28}$ | $-16.25_{\pm0.28}$ | $-16.25_{\pm0.28}$ | $-16.25_{\pm0.28}$ |
| **Task** $f(x^*_{OFF})$ | CEC 3 $-2.0_{\pm0.00}$ | | | | | | | | | | | | | | |
| CARCOO | $-2.06_{\pm0.03}$ | $-3.43_{\pm2.20}$ | $-3.42_{\pm2.21}$ | $-3.43_{\pm2.20}$ | $-3.42_{\pm2.20}$ | $-2.07_{\pm0.06}$ | $-4.16_{\pm3.15}$ | $-4.15_{\pm3.16}$ | $-4.16_{\pm3.15}$ | $-4.16_{\pm3.15}$ | $-2.15_{\pm0.25}$ | $-7.34_{\pm10.16}$ | $-7.33_{\pm10.16}$ | $-7.33_{\pm10.16}$ | $-7.34_{\pm10.16}$ |
| DEPF | $-2.62_{\pm0.29}$ | $-40.62_{\pm11.99}$ | $-40.79_{\pm11.87}$ | $-40.71_{\pm11.95}$ | $-40.73_{\pm11.91}$ | $-2.79_{\pm0.02}$ | $-58.1_{\pm7.90}$ | $-58.40_{\pm7.79}$ | $-58.25_{\pm7.84}$ | $-58.20_{\pm7.86}$ | $-2.79_{\pm0.02}$ | $-64.59_{\pm5.61}$ | $-64.92_{\pm5.49}$ | $-64.75_{\pm5.53}$ | $-64.59_{\pm5.61}$ |
| DESPF | $-2.68_{\pm0.25}$ | $-37.74_{\pm18.15}$ | $-38.66_{\pm18.19}$ | $-38.16_{\pm18.13}$ | $-38.32_{\pm18.14}$ | $-2.70_{\pm0.25}$ | $-47.81_{\pm17.82}$ | $-49.13_{\pm18.36}$ | $-48.41_{\pm17.98}$ | $-48.20_{\pm17.91}$ | $-2.71_{\pm0.22}$ | $-54.51_{\pm17.28}$ | $-55.97_{\pm17.80}$ | $-55.17_{\pm17.41}$ | $-54.52_{\pm17.28}$ |
| PRIME | $-2.65_{\pm0.27}$ | $-72.73_{\pm3.38}$ | $-72.73_{\pm3.38}$ | $-72.73_{\pm3.38}$ | $-72.73_{\pm3.38}$ | $-2.79_{\pm0.02}$ | $-70.75_{\pm3.73}$ | $-70.75_{\pm3.73}$ | $-70.75_{\pm3.73}$ | $-70.75_{\pm3.73}$ | $-2.79_{\pm0.02}$ | $-73.06_{\pm2.75}$ | $-73.06_{\pm2.75}$ | $-73.06_{\pm2.75}$ | $-73.06_{\pm2.75}$ |
| **Task** $f(x^*_{OFF})$ | CEC 4 $-264.43_{\pm0.30}$ | | | | | | | | | | | | | | |
| CARCOO | $-286.68_{\pm0.71}$ | $-2.20e3_{\pm71.44}$ | $-2.20e3_{\pm71.44}$ | $-2.20e3_{\pm71.44}$ | $-2.20e3_{\pm71.44}$ | $-286.68_{\pm0.71}$ | $-2.21e3_{\pm71.82}$ | $-2.21e3_{\pm71.82}$ | $-2.21e3_{\pm71.82}$ | $-2.21e3_{\pm71.82}$ | $-286.68_{\pm0.71}$ | $-2.22e3_{\pm71.94}$ | $-2.22e3_{\pm71.94}$ | $-2.22e3_{\pm71.94}$ | $-2.22e3_{\pm71.94}$ |
| DEPF | $-286.68_{\pm0.71}$ | $-1.69e3_{\pm128.24}$ | $-1.69e3_{\pm128.24}$ | $-1.69e3_{\pm128.24}$ | $-1.69e3_{\pm128.24}$ | $-291.88_{\pm9.31}$ | $-2.08e3_{\pm264.34}$ | $-2.08e3_{\pm264.34}$ | $-2.08e3_{\pm264.34}$ | $-2.08e3_{\pm264.34}$ | $-304.75_{\pm16.43}$ | $-2.59e3_{\pm523.61}$ | $-2.59e3_{\pm523.61}$ | $-2.59e3_{\pm523.61}$ | $-2.59e3_{\pm523.61}$ |
| DESPF | $-286.56_{\pm0.84}$ | $-1.48e3_{\pm571.10}$ | $-1.49e3_{\pm562.74}$ | $-1.48e3_{\pm568.31}$ | $-1.48e3_{\pm566.90}$ | $-302.79_{\pm11.55}$ | $-1.89e3_{\pm699.98}$ | $-1.90e3_{\pm681.50}$ | $-1.89e3_{\pm693.82}$ | $-1.89e3_{\pm696.22}$ | $-316.87_{\pm20.62}$ | $-2.49e3_{\pm1.04e3}$ | $-2.50e3_{\pm1.01e3}$ | $-2.49e3_{\pm1.03e3}$ | $-2.49e3_{\pm1.04e3}$ |
| PRIME | $-286.68_{\pm0.71}$ | $-2.20e3_{\pm71.44}$ | $-2.20e3_{\pm71.44}$ | $-2.20e3_{\pm71.44}$ | $-2.20e3_{\pm71.44}$ | $-286.68_{\pm0.71}$ | $-2.21e3_{\pm72.36}$ | $-2.21e3_{\pm72.36}$ | $-2.21e3_{\pm72.36}$ | $-2.21e3_{\pm72.36}$ | $-286.68_{\pm0.71}$ | $-2.21e3_{\pm71.78}$ | $-2.21e3_{\pm71.78}$ | $-2.21e3_{\pm71.78}$ | $-2.21e3_{\pm71.78}$ |
| **Task** $f(x^*_{OFF})$ | CEC 5 $-4.91_{\pm0.10}$ | | | | | | | | | | | | | | |
| CARCOO | $-16.07_{\pm0.24}$ | $-1.51_{\pm0.18}$ | $-3.02_{\pm0.35}$ | $-2.01_{\pm0.23}$ | $-2.27_{\pm0.26}$ | $-16.07_{\pm0.24}$ | $-1.58_{\pm0.18}$ | $-3.15_{\pm0.37}$ | $-2.10_{\pm0.25}$ | $-1.90_{\pm0.22}$ | $-16.07_{\pm0.24}$ | $-1.60_{\pm0.19}$ | $-3.20_{\pm0.38}$ | $-2.13_{\pm0.25}$ | $-1.60_{\pm0.19}$ |
| DEPF | $-13.13_{\pm2.87}$ | $-52.29_{\pm133.88}$ | $-53.78_{\pm133.38}$ | $-52.79_{\pm133.71}$ | $-53.04_{\pm133.63}$ | $-15.97_{\pm40.40}$ | $-84.75_{\pm226.06}$ | $-87.45_{\pm215.11}$ | $-85.65_{\pm215.74}$ | $-85.29_{\pm215.87}$ | $-16.08_{\pm0.23}$ | $-106.13_{\pm271.17}$ | $-109.31_{\pm270.70}$ | $-107.19_{\pm270.79}$ | $-106.14_{\pm271.17}$ |
| DESPF | $-16.08_{\pm0.23}$ | $-124.72_{\pm330.56}$ | $-131.15_{\pm308.22}$ | $-126.86_{\pm309.78}$ | $-127.95_{\pm309.38}$ | $-16.08_{\pm0.23}$ | $-137.64_{\pm341.81}$ | $-145.04_{\pm339.11}$ | $-140.10_{\pm340.90}$ | $-139.14_{\pm341.25}$ | $-15.99_{\pm0.25}$ | $-142.17_{\pm352.86}$ | $-149.88_{\pm350.05}$ | $-144.74_{\pm351.91}$ | $-142.19_{\pm352.85}$ |
| PRIME | $-16.08_{\pm0.23}$ | $-2.11_{\pm0.39}$ | $-4.22_{\pm0.79}$ | $-2.81_{\pm0.53}$ | $-3.17_{\pm0.59}$ | $-16.08_{\pm0.23}$ | $-2.20_{\pm0.40}$ | $-4.41_{\pm0.79}$ | $-2.94_{\pm0.53}$ | $-2.65_{\pm0.48}$ | $-16.08_{\pm0.23}$ | $-2.23_{\pm0.40}$ | $-4.47_{\pm0.79}$ | $-2.98_{\pm0.53}$ | $-2.24_{\pm0.40}$ |

Table 73: Overall results for CEC constrained tasks with 128 solutions and 100th percentile evaluations. In this case, 40% of the values are missing near the worst value and another 10% near the optimal value. Details are the same as Table 5.

| Steps | t = 50 | | | | | t = 100 | | | | | t = 150 | | | | |
|---|---|---|---|---|---|---|---|---|---|---|---|---|---|---|---|
| **Task** $f(x^*_{OFF})$ | CEC 1 $-8.10e3_{\pm108.86}$ | | | | | | | | | | | | | | |
| Metric | FS | SI | OI | SO | SO$_\omega$ | FS | SI | OI | SO | SO$_\omega$ | FS | SI | OI | SO | SO$_\omega$ |
| CARCOO | - | - | - | - | - | - | - | - | - | - | - | - | - | - | - |
| DEPF | $-8.10e3_{\pm108.86}$ | $1.00_{\pm0.00}$ | $1.00_{\pm0.00}$ | $1.00_{\pm0.00}$ | $1.00_{\pm0.00}$ | $-8.10e3_{\pm108.86}$ | $1.00_{\pm0.00}$ | $1.00_{\pm0.00}$ | $1.00_{\pm0.00}$ | $1.00_{\pm0.00}$ | $-8.10e3_{\pm108.86}$ | $1.00_{\pm0.00}$ | $1.00_{\pm0.00}$ | $1.00_{\pm0.00}$ | $1.00_{\pm0.00}$ |
| DESPF | $-8.10e3_{\pm108.86}$ | $1.00_{\pm0.00}$ | $1.00_{\pm0.00}$ | $1.00_{\pm0.00}$ | $1.00_{\pm0.00}$ | $-8.10e3_{\pm108.86}$ | $1.00_{\pm0.00}$ | $1.00_{\pm0.00}$ | $1.00_{\pm0.00}$ | $1.00_{\pm0.00}$ | $-8.10e3_{\pm108.86}$ | $1.00_{\pm0.00}$ | $1.00_{\pm0.00}$ | $1.00_{\pm0.00}$ | $1.00_{\pm0.00}$ |
| PRIME | - | - | - | - | - | - | - | - | - | - | - | - | - | - | - |
| **Task** $f(x^*_{OFF})$ | CEC 2 $-1.08_{\pm0.00}$ | | | | | | | | | | | | | | |
| CARCOO | $-1.21_{\pm0.02}$ | $-11.16_{\pm0.83}$ | $-11.16_{\pm0.83}$ | $-11.16_{\pm0.83}$ | $-11.16_{\pm0.83}$ | $-1.22_{\pm0.03}$ | $-11.80_{\pm1.80}$ | $-11.80_{\pm1.80}$ | $-11.80_{\pm1.80}$ | $-11.80_{\pm1.80}$ | $-1.22_{\pm0.04}$ | $-12.06_{\pm2.35}$ | $-12.06_{\pm2.35}$ | $-12.06_{\pm2.35}$ | $-12.06_{\pm2.35}$ |
| DEPF | $-1.21_{\pm0.00}$ | $-11.18_{\pm0.77}$ | $-11.18_{\pm0.77}$ | $-11.18_{\pm0.77}$ | $-11.18_{\pm0.77}$ | $-1.21_{\pm0.01}$ | $-11.56_{\pm0.82}$ | $-11.56_{\pm0.82}$ | $-11.56_{\pm0.82}$ | $-11.56_{\pm0.82}$ | $-1.21_{\pm0.01}$ | $-11.76_{\pm1.03}$ | $-11.76_{\pm1.03}$ | $-11.76_{\pm1.03}$ | $-11.76_{\pm1.03}$ |
| DESPF | $-1.21_{\pm0.00}$ | $-11.48_{\pm0.42}$ | $-11.48_{\pm0.42}$ | $-11.48_{\pm0.42}$ | $-11.48_{\pm0.42}$ | $-1.21_{\pm0.00}$ | $-11.55_{\pm0.41}$ | $-11.55_{\pm0.41}$ | $-11.55_{\pm0.41}$ | $-11.55_{\pm0.41}$ | $-1.21_{\pm0.00}$ | $-11.58_{\pm0.41}$ | $-11.58_{\pm0.41}$ | $-11.58_{\pm0.41}$ | $-11.58_{\pm0.41}$ |
| PRIME | $-1.21_{\pm0.00}$ | $-11.33_{\pm0.53}$ | $-11.33_{\pm0.53}$ | $-11.33_{\pm0.53}$ | $-11.33_{\pm0.53}$ | $-1.21_{\pm0.00}$ | $-11.48_{\pm0.46}$ | $-11.48_{\pm0.46}$ | $-11.48_{\pm0.46}$ | $-11.48_{\pm0.46}$ | $-1.21_{\pm0.00}$ | $-11.53_{\pm0.44}$ | $-11.53_{\pm0.44}$ | $-11.53_{\pm0.44}$ | $-11.53_{\pm0.44}$ |
| **Task** $f(x^*_{OFF})$ | CEC 3 $-2.0_{\pm0.00}$ | | | | | | | | | | | | | | |
| CARCOO | $-2.09_{\pm0.05}$ | $-6.93_{\pm4.30}$ | $-6.93_{\pm4.30}$ | $-6.93_{\pm4.30}$ | $-6.93_{\pm4.30}$ | $-2.15_{\pm0.18}$ | $-9.80_{\pm6.63}$ | $-9.80_{\pm6.63}$ | $-9.80_{\pm6.63}$ | $-9.80_{\pm6.63}$ | $-2.15_{\pm0.18}$ | $-11.28_{\pm11.66}$ | $-11.28_{\pm11.66}$ | $-11.28_{\pm11.66}$ | $-11.28_{\pm11.66}$ |
| DEPF | $-2.49_{\pm0.17}$ | $-28.42_{\pm8.34}$ | $-28.50_{\pm8.21}$ | $-28.46_{\pm8.28}$ | $-28.47_{\pm8.25}$ | $-2.44_{\pm0.22}$ | $-37.52_{\pm7.19}$ | $-37.60_{\pm7.10}$ | $-37.58_{\pm7.13}$ | $-37.60_{\pm7.10}$ | $-2.50_{\pm0.17}$ | $-39.68_{\pm9.04}$ | $-39.88_{\pm9.00}$ | $-39.78_{\pm9.02}$ | $-39.68_{\pm9.04}$ |
| DESPF | $-2.52_{\pm0.05}$ | $-21.38_{\pm6.17}$ | $-21.81_{\pm5.53}$ | $-21.58_{\pm5.87}$ | $-21.65_{\pm5.76}$ | $-2.56_{\pm0.02}$ | $-35.48_{\pm6.07}$ | $-36.39_{\pm4.68}$ | $-35.90_{\pm5.41}$ | $-35.76_{\pm5.63}$ | $-2.50_{\pm0.16}$ | $-41.52_{\pm5.25}$ | $-42.69_{\pm3.29}$ | $-42.06_{\pm4.33}$ | $-41.53_{\pm5.24}$ |
| PRIME | $-2.51_{\pm0.14}$ | $-54.16_{\pm2.36}$ | $-54.16_{\pm2.36}$ | $-54.16_{\pm2.36}$ | $-54.16_{\pm2.36}$ | $-2.57_{\pm0.02}$ | $-51.74_{\pm2.58}$ | $-51.84_{\pm2.50}$ | $-51.79_{\pm2.54}$ | $-51.78_{\pm2.55}$ | $-2.57_{\pm0.02}$ | $-52.97_{\pm2.13}$ | $-53.08_{\pm2.09}$ | $-53.02_{\pm2.10}$ | $-52.97_{\pm2.13}$ |
| **Task** $f(x^*_{OFF})$ | CEC 4 $-264.33_{\pm0.20}$ | | | | | | | | | | | | | | |
| CARCOO | - | - | - | - | - | - | - | - | - | - | - | - | - | - | - |
| DEPF | $-268.48_{\pm1.33}$ | $-177.19_{\pm188.60}$ | $-178.25_{\pm187.65}$ | $-177.57_{\pm188.29}$ | $-177.75_{\pm188.09}$ | $-277.65_{\pm12.77}$ | $-345.18_{\pm407.75}$ | $-348.23_{\pm405.28}$ | $-346.21_{\pm406.91}$ | $-345.81_{\pm407.23}$ | $-283.67_{\pm15.36}$ | $-590.49_{\pm674.85}$ | $-595.79_{\pm670.37}$ | $-592.26_{\pm673.35}$ | $-590.57_{\pm674.83}$ |
| DESPF | $-273.73_{\pm12.26}$ | $-569.57_{\pm354.67}$ | $-569.57_{\pm354.67}$ | $-569.57_{\pm354.67}$ | $-569.57_{\pm354.67}$ | $-297.69_{\pm16.88}$ | $-1.30e3_{\pm887.41}$ | $-1.30e3_{\pm887.41}$ | $-1.30e3_{\pm887.41}$ | $-1.30e3_{\pm887.41}$ | $-322.82_{\pm23.30}$ | $-2.38e3_{\pm1.15e3}$ | $-2.38e3_{\pm1.15e3}$ | $-2.38e3_{\pm1.15e3}$ | $-2.38e3_{\pm1.15e3}$ |
| PRIME | $-269.17_{\pm0.88}$ | $-477.08_{\pm97.49}$ | $-477.08_{\pm97.49}$ | $-477.08_{\pm97.49}$ | $-477.08_{\pm97.49}$ | $-269.17_{\pm0.88}$ | $-479.57_{\pm98.00}$ | $-479.57_{\pm98.00}$ | $-479.57_{\pm98.00}$ | $-479.57_{\pm98.00}$ | $-269.17_{\pm0.88}$ | $-480.39_{\pm98.16}$ | $-480.39_{\pm98.16}$ | $-480.39_{\pm98.16}$ | $-480.39_{\pm98.16}$ |
| **Task** $f(x^*_{OFF})$ | CEC 5 $-2.95_{\pm0.00}$ | | | | | | | | | | | | | | |
| CARCOO | $-13.06_{\pm0.19}$ | $-5.54_{\pm1.10}$ | $-11.08_{\pm2.20}$ | $-7.39_{\pm1.47}$ | $-8.33_{\pm1.65}$ | $-13.06_{\pm0.19}$ | $-5.67_{\pm1.11}$ | $-11.34_{\pm2.23}$ | $-7.56_{\pm1.48}$ | $-6.82_{\pm1.34}$ | $-13.06_{\pm0.19}$ | $-5.71_{\pm1.12}$ | $-11.42_{\pm2.23}$ | $-7.61_{\pm1.49}$ | $-5.73_{\pm1.12}$ |
| DEPF | $-10.84_{\pm1.71}$ | $-324.10_{\pm161.04}$ | $-337.20_{\pm145.76}$ | $-328.46_{\pm155.29}$ | $-330.68_{\pm152.61}$ | $-13.06_{\pm0.19}$ | $-515.92_{\pm260.10}$ | $-535.50_{\pm236.55}$ | $-522.45_{\pm251.36}$ | $-519.90_{\pm254.67}$ | $-13.06_{\pm0.19}$ | $-605.03_{\pm303.33}$ | $-627.06_{\pm275.74}$ | $-612.37_{\pm293.20}$ | $-605.11_{\pm303.22}$ |
| DESPF | $-13.06_{\pm0.19}$ | $-749.44_{\pm284.09}$ | $-750.65_{\pm280.94}$ | $-749.84_{\pm283.04}$ | $-750.04_{\pm282.50}$ | $-13.06_{\pm0.19}$ | $-819.21_{\pm307.05}$ | $-820.55_{\pm303.45}$ | $-819.67_{\pm305.85}$ | $-819.49_{\pm306.32}$ | $-13.06_{\pm0.19}$ | $-842.16_{\pm315.02}$ | $-843.58_{\pm311.27}$ | $-842.63_{\pm313.77}$ | $-842.16_{\pm315.00}$ |
| PRIME | $-13.06_{\pm0.19}$ | $-503.92_{\pm494.78}$ | $-508.62_{\pm490.09}$ | $-505.49_{\pm493.21}$ | $-506.28_{\pm492.42}$ | $-13.06_{\pm0.19}$ | $-506.60_{\pm497.30}$ | $-511.39_{\pm492.53}$ | $-508.20_{\pm495.71}$ | $-507.57_{\pm496.53}$ | $-13.06_{\pm0.19}$ | $-507.48_{\pm498.13}$ | $-512.30_{\pm493.33}$ | $-509.09_{\pm496.53}$ | $-507.50_{\pm498.11}$ |

Table 74: Overall results for CEC constrained tasks with 128 solutions and 50th percentile evaluations. In this case, 0% of the values are missing near the worst value and another 50% near the optimal value. Details are the same as Table 5.

| Steps | t = 50 | | | | | t = 100 | | | | | t = 150 | | | | |
|---|---|---|---|---|---|---|---|---|---|---|---|---|---|---|---|
| **Task** $f(x^*_{OFF})$ | CEC 1 $-2.08e4_{\pm424.97}$ | | | | | | | | | | | | | | |
| Metric | FS | SI | OI | SO | SO$_\omega$ | FS | SI | OI | SO | SO$_\omega$ | FS | SI | OI | SO | SO$_\omega$ |
| CARCOO | - | - | - | - | - | - | - | - | - | - | - | - | - | - | - |
| DEPF | $-2.08e4_{\pm424.97}$ | $1.00_{\pm0.00}$ | $1.00_{\pm0.00}$ | $1.00_{\pm0.00}$ | $1.00_{\pm0.00}$ | $-2.08e4_{\pm424.97}$ | $1.00_{\pm0.00}$ | $1.00_{\pm0.00}$ | $1.00_{\pm0.00}$ | $1.00_{\pm0.00}$ | $-2.08e4_{\pm424.97}$ | $1.00_{\pm0.00}$ | $1.00_{\pm0.00}$ | $1.00_{\pm0.00}$ | $1.00_{\pm0.00}$ |
| DESPF | $-2.08e4_{\pm424.97}$ | $1.00_{\pm0.00}$ | $1.00_{\pm0.00}$ | $1.00_{\pm0.00}$ | $1.00_{\pm0.00}$ | $-2.08e4_{\pm424.97}$ | $1.00_{\pm0.00}$ | $1.00_{\pm0.00}$ | $1.00_{\pm0.00}$ | $1.00_{\pm0.00}$ | $-2.08e4_{\pm424.97}$ | $1.00_{\pm0.00}$ | $1.00_{\pm0.00}$ | $1.00_{\pm0.00}$ | $1.00_{\pm0.00}$ |
| PRIME | - | - | - | - | - | - | - | - | - | - | - | - | - | - | - |
| **Task** $f(x^*_{OFF})$ | CEC 2 $-1.14_{\pm0.01}$ | | | | | | | | | | | | | | |
| CARCOO | $-2.05_{\pm0.00}$ | $-70.97_{\pm28.99}$ | $-70.97_{\pm28.99}$ | $-70.97_{\pm28.99}$ | $-70.97_{\pm28.99}$ | $-2.05_{\pm0.00}$ | $-71.37_{\pm29.10}$ | $-71.37_{\pm29.10}$ | $-71.37_{\pm29.10}$ | $-71.37_{\pm29.10}$ | $-2.05_{\pm0.00}$ | $-71.50_{\pm29.13}$ | $-71.50_{\pm29.13}$ | $-71.50_{\pm29.13}$ | $-71.50_{\pm29.13}$ |
| DEPF | $-1.69_{\pm0.36}$ | $-64.74_{\pm20.58}$ | $-64.74_{\pm20.58}$ | $-64.74_{\pm20.58}$ | $-64.74_{\pm20.58}$ | $-1.86_{\pm0.32}$ | $-61.03_{\pm24.64}$ | $-61.03_{\pm24.64}$ | $-61.03_{\pm24.64}$ | $-61.03_{\pm24.64}$ | $-1.87_{\pm0.30}$ | $-62.28_{\pm26.61}$ | $-62.28_{\pm26.61}$ | $-62.28_{\pm26.61}$ | $-62.28_{\pm26.61}$ |
| DESPF | $-2.04_{\pm0.00}$ | $-88.81_{\pm0.74}$ | $-88.81_{\pm0.74}$ | $-88.81_{\pm0.74}$ | $-88.81_{\pm0.74}$ | $-2.04_{\pm0.00}$ | $-89.28_{\pm0.75}$ | $-89.28_{\pm0.75}$ | $-89.28_{\pm0.75}$ | $-89.28_{\pm0.75}$ | $-2.04_{\pm0.00}$ | $-89.44_{\pm0.75}$ | $-89.44_{\pm0.75}$ | $-89.44_{\pm0.75}$ | $-89.44_{\pm0.75}$ |
| PRIME | $-2.04_{\pm0.00}$ | $-16.53_{\pm2.15}$ | $-16.53_{\pm2.15}$ | $-16.53_{\pm2.15}$ | $-16.53_{\pm2.15}$ | $-2.04_{\pm0.00}$ | $-16.74_{\pm2.19}$ | $-16.74_{\pm2.19}$ | $-16.74_{\pm2.19}$ | $-16.74_{\pm2.19}$ | $-2.04_{\pm0.00}$ | $-16.81_{\pm2.20}$ | $-16.81_{\pm2.20}$ | $-16.81_{\pm2.20}$ | $-16.81_{\pm2.20}$ |
| **Task** $f(x^*_{OFF})$ | CEC 3 $-2.35_{\pm0.02}$ | | | | | | | | | | | | | | |
| CARCOO | $-3.96_{\pm0.65}$ | $-7.94_{\pm4.14}$ | $-11.32_{\pm5.94}$ | $-9.32_{\pm4.87}$ | $-9.91_{\pm5.18}$ | $-3.96_{\pm0.63}$ | $-8.56_{\pm4.33}$ | $-12.24_{\pm6.21}$ | $-10.06_{\pm5.08}$ | $-9.52_{\pm4.80}$ | $-3.96_{\pm0.63}$ | $-8.77_{\pm4.40}$ | $-12.54_{\pm6.31}$ | $-10.30_{\pm5.16}$ | $-8.78_{\pm4.41}$ |
| DEPF | $-4.20_{\pm0.00}$ | $-4.66_{\pm0.69}$ | $-4.66_{\pm0.69}$ | $-4.66_{\pm0.69}$ | $-4.66_{\pm0.69}$ | $-4.20_{\pm0.00}$ | $-4.69_{\pm0.70}$ | $-4.69_{\pm0.70}$ | $-4.69_{\pm0.70}$ | $-4.69_{\pm0.70}$ | $-4.20_{\pm0.00}$ | $-4.70_{\pm0.70}$ | $-4.70_{\pm0.70}$ | $-4.70_{\pm0.70}$ | $-4.70_{\pm0.70}$ |
| DESPF | $-4.20_{\pm0.00}$ | $-4.57_{\pm0.76}$ | $-4.57_{\pm0.76}$ | $-4.57_{\pm0.76}$ | $-4.57_{\pm0.76}$ | $-4.20_{\pm0.00}$ | $-4.60_{\pm0.77}$ | $-4.60_{\pm0.77}$ | $-4.60_{\pm0.77}$ | $-4.60_{\pm0.77}$ | $-4.20_{\pm0.00}$ | $-4.61_{\pm0.77}$ | $-4.61_{\pm0.77}$ | $-4.61_{\pm0.77}$ | $-4.61_{\pm0.77}$ |
| PRIME | $-4.20_{\pm0.00}$ | $-3.90_{\pm0.50}$ | $-4.11_{\pm0.58}$ | $-4.00_{\pm0.44}$ | $-4.04_{\pm0.42}$ | $-4.20_{\pm0.00}$ | $-4.00_{\pm0.49}$ | $-4.22_{\pm0.57}$ | $-4.11_{\pm0.43}$ | $-4.07_{\pm0.45}$ | $-4.20_{\pm0.00}$ | $-4.04_{\pm0.49}$ | $-4.26_{\pm0.56}$ | $-4.14_{\pm0.43}$ | $-4.04_{\pm0.49}$ |
| **Task** $f(x^*_{OFF})$ | CEC 4 $-264.43_{\pm0.30}$ | | | | | | | | | | | | | | |
| CARCOO | $-379.59_{\pm1.78}$ | $-1.14e4_{\pm195.53}$ | $-1.14e4_{\pm195.53}$ | $-1.14e4_{\pm195.53}$ | $-1.14e4_{\pm195.53}$ | $-379.59_{\pm1.78}$ | $-1.15e4_{\pm196.54}$ | $-1.15e4_{\pm196.54}$ | $-1.15e4_{\pm196.54}$ | $-1.15e4_{\pm196.54}$ | $-379.59_{\pm1.78}$ | $-1.15e4_{\pm196.88}$ | $-1.15e4_{\pm196.88}$ | $-1.15e4_{\pm196.88}$ | $-1.15e4_{\pm196.88}$ |
| DEPF | $-382.84_{\pm0.00}$ | $-1.14e4_{\pm126.18}$ | $-1.14e4_{\pm126.18}$ | $-1.14e4_{\pm126.18}$ | $-1.14e4_{\pm126.18}$ | $-382.84_{\pm0.00}$ | $-1.16e4_{\pm75.96}$ | $-1.16e4_{\pm75.96}$ | $-1.16e4_{\pm75.96}$ | $-1.16e4_{\pm75.96}$ | $-382.84_{\pm0.00}$ | $-1.17e4_{\pm59.82}$ | $-1.17e4_{\pm59.82}$ | $-1.17e4_{\pm59.82}$ | $-1.17e4_{\pm59.82}$ |
| DESPF | $-382.57_{\pm0.37}$ | $-1.12e4_{\pm320.38}$ | $-1.12e4_{\pm320.38}$ | $-1.12e4_{\pm320.38}$ | $-1.12e4_{\pm320.38}$ | $-382.84_{\pm0.01}$ | $-1.15e4_{\pm164.93}$ | $-1.15e4_{\pm164.93}$ | $-1.15e4_{\pm164.93}$ | $-1.15e4_{\pm164.93}$ | $-382.84_{\pm0.00}$ | $-1.16e4_{\pm112.18}$ | $-1.16e4_{\pm112.18}$ | $-1.16e4_{\pm112.18}$ | $-1.16e4_{\pm112.18}$ |
| PRIME | $-379.59_{\pm1.78}$ | $-1.14e4_{\pm195.53}$ | $-1.14e4_{\pm195.53}$ | $-1.14e4_{\pm195.53}$ | $-1.14e4_{\pm195.53}$ | $-379.59_{\pm1.76}$ | $-1.15e4_{\pm196.54}$ | $-1.15e4_{\pm196.54}$ | $-1.15e4_{\pm196.54}$ | $-1.15e4_{\pm196.54}$ | $-379.59_{\pm1.78}$ | $-1.15e4_{\pm196.88}$ | $-1.15e4_{\pm196.88}$ | $-1.15e4_{\pm196.88}$ | $-1.15e4_{\pm196.88}$ |
| **Task** $f(x^*_{OFF})$ | CEC 5 $-10.71_{\pm0.18}$ | | | | | | | | | | | | | | |
| CARCOO | $-58.62_{\pm3.36}$ | $-3.51_{\pm0.32}$ | $-4.48_{\pm0.35}$ | $-3.94_{\pm0.33}$ | $-4.11_{\pm0.34}$ | $-58.62_{\pm3.36}$ | $-3.63_{\pm0.33}$ | $-4.64_{\pm0.36}$ | $-4.07_{\pm0.34}$ | $-3.92_{\pm0.34}$ | $-58.62_{\pm3.36}$ | $-3.67_{\pm0.33}$ | $-4.69_{\pm0.36}$ | $-4.12_{\pm0.34}$ | $-3.68_{\pm0.33}$ |
| DEPF | $-59.28_{\pm3.99}$ | $-5.70_{\pm0.96}$ | $-5.70_{\pm0.96}$ | $-5.70_{\pm0.96}$ | $-5.70_{\pm0.96}$ | $-61.08_{\pm4.75}$ | $-5.84_{\pm0.95}$ | $-5.84_{\pm0.95}$ | $-5.84_{\pm0.95}$ | $-5.84_{\pm0.95}$ | $-62.16_{\pm5.65}$ | $-5.95_{\pm0.95}$ | $-5.95_{\pm0.95}$ | $-5.95_{\pm0.95}$ | $-5.95_{\pm0.95}$ |
| DESPF | $-67.28_{\pm0.00}$ | $-8.40_{\pm1.93}$ | $-8.40_{\pm1.93}$ | $-8.40_{\pm1.93}$ | $-8.40_{\pm1.93}$ | $-67.28_{\pm0.00}$ | $-8.80_{\pm1.90}$ | $-8.80_{\pm1.90}$ | $-8.80_{\pm1.90}$ | $-8.80_{\pm1.90}$ | $-67.28_{\pm0.00}$ | $-8.94_{\pm1.89}$ | $-8.94_{\pm1.89}$ | $-8.94_{\pm1.89}$ | $-8.94_{\pm1.89}$ |
| PRIME | $-58.62_{\pm3.36}$ | $-5.28_{\pm0.46}$ | $-5.30_{\pm0.48}$ | $-5.29_{\pm0.47}$ | $-5.29_{\pm0.47}$ | $-58.62_{\pm3.36}$ | $-5.33_{\pm0.47}$ | $-5.34_{\pm0.49}$ | $-5.33_{\pm0.48}$ | $-5.33_{\pm0.48}$ | $-58.62_{\pm3.36}$ | $-5.34_{\pm0.48}$ | $-5.35_{\pm0.49}$ | $-5.35_{\pm0.48}$ | $-5.34_{\pm0.48}$ |

Table 75: Overall results for CEC constrained tasks with 128 solutions and 50th percentile evaluations. In this case, 0% of the values are missing near the worst value and another 40% near the optimal value. Details are the same as Table 5.

| Steps | t = 50 | | | | | t = 100 | | | | | t = 150 | | | | |
|---|---|---|---|---|---|---|---|---|---|---|---|---|---|---|---|
| Task f(x*_OFF) | | | | | | CEC 1  −2.08e4±424.97 | | | | | | | | | |
| Metric | FS | SI | OI | SO | SO_ω | FS | SI | OI | SO | SO_ω | FS | SI | OI | SO | SO_ω |
| CARCOO | - | - | - | - | - | - | - | - | - | - | - | - | - | - | - |
| DEPF | −2.08e4±424.97 | 1.00±.00 | 1.00±.00 | 1.00±.00 | 1.00±.00 | −2.08e4±424.97 | 1.00±.00 | 1.00±.00 | 1.00±.00 | 1.00±.00 | −2.08e4±424.97 | 1.00±.00 | 1.00±.00 | 1.00±.00 | 1.00±.00 |
| DESPF | −2.08e4±424.97 | 1.00±.00 | 1.00±.00 | 1.00±.00 | 1.00±.00 | −2.08e4±424.97 | 1.00±.00 | 1.00±.00 | 1.00±.00 | 1.00±.00 | −2.08e4±424.97 | 1.00±.00 | 1.00±.00 | 1.00±.00 | 1.00±.00 |
| PRIME | - | - | - | - | - | - | - | - | - | - | - | - | - | - | - |
| Task f(x*_OFF) | | | | | | CEC 2  −1.08±.00 | | | | | | | | | |
| CARCOO | −2.04±.00 | −94.28±.62 | −94.28±.62 | −94.28±.62 | −94.28±.62 | −2.04±.00 | −94.78±.63 | −94.78±.63 | −94.78±.63 | −94.78±.63 | −2.04±.00 | −94.94±.63 | −94.94±.63 | −94.94±.63 | −94.94±.63 |
| DEPF | −1.96±.23 | −89.28±12.89 | −89.28±12.89 | −89.28±12.89 | −89.28±12.89 | −1.85±.14 | −80.76±18.55 | −80.76±18.55 | −80.76±18.55 | −80.76±18.55 | −1.95±.26 | −80.46±19.48 | −80.46±19.48 | −80.46±19.48 | −80.46±19.48 |
| DESPF | −2.04±.00 | −94.28±.62 | −94.28±.62 | −94.28±.62 | −94.28±.62 | −2.04±.00 | −94.78±.63 | −94.78±.63 | −94.78±.63 | −94.78±.63 | −2.04±.00 | −94.94±.63 | −94.94±.63 | −94.94±.63 | −94.94±.63 |
| PRIME | −2.04±.00 | −94.28±.62 | −94.28±.62 | −94.28±.62 | −94.28±.62 | −2.04±.00 | −94.78±.63 | −94.78±.63 | −94.78±.63 | −94.78±.63 | −2.04±.00 | −94.94±.63 | −94.94±.63 | −94.94±.63 | −94.94±.63 |
| Task f(x*_OFF) | | | | | | CEC 3  −2.12±.02 | | | | | | | | | |
| CARCOO | −4.20±.00 | −18.33±2.67 | −18.33±2.67 | −18.33±2.67 | −18.33±2.67 | −4.20±.00 | −18.43±2.69 | −18.43±2.69 | −18.43±2.69 | −18.43±2.69 | −4.20±.00 | −18.46±2.69 | −18.46±2.69 | −18.46±2.69 | −18.46±2.69 |
| DEPF | −4.20±.00 | −17.64±2.82 | −17.64±2.82 | −17.64±2.82 | −17.64±2.82 | −4.20±.00 | −17.73±2.83 | −17.73±2.83 | −17.73±2.83 | −17.73±2.83 | −4.20±.00 | −17.77±2.84 | −17.77±2.84 | −17.77±2.84 | −17.77±2.84 |
| DESPF | −4.20±.00 | −68.73±78.55 | −68.73±78.55 | −68.73±78.55 | −68.73±78.55 | −4.20±.00 | −69.09±78.94 | −69.09±78.94 | −69.09±78.94 | −69.09±78.94 | −4.20±.00 | −69.21±79.07 | −69.21±79.07 | −69.21±79.07 | −69.21±79.07 |
| PRIME | −4.20±.00 | −16.53±2.31 | −16.53±2.31 | −16.53±2.31 | −16.53±2.31 | −4.20±.00 | −16.68±2.35 | −16.68±2.35 | −16.68±2.35 | −16.68±2.35 | −4.20±.00 | −16.73±2.37 | −16.73±2.37 | −16.73±2.37 | −16.73±2.37 |
| Task f(x*_OFF) | | | | | | CEC 4  −264.43±.30 | | | | | | | | | |
| CARCOO | −379.59±1.78 | −1.14e4±195.53 | −1.14e4±195.53 | −1.14e4±195.53 | −1.14e4±195.53 | −379.59±1.78 | −1.15e4±196.54 | −1.15e4±196.54 | −1.15e4±196.54 | −1.15e4±196.54 | −379.59±1.78 | −1.15e4±196.88 | −1.15e4±196.88 | −1.15e4±196.88 | −1.15e4±196.88 |
| DEPF | −382.84±.00 | −8.85e3±4.74e3 | −8.85e3±4.74e3 | −8.85e3±4.74e3 | −8.85e3±4.74e3 | −382.84±.00 | −8.95e3±4.79e3 | −8.95e3±4.79e3 | −8.95e3±4.79e3 | −8.95e3±4.79e3 | −382.84±.00 | −8.99e3±4.83e3 | −8.99e3±4.83e3 | −8.99e3±4.83e3 | −8.99e3±4.83e3 |
| DESPF | −382.61±.34 | −1.00e4±3.52e3 | −1.00e4±3.52e3 | −1.00e4±3.52e3 | −1.00e4±3.52e3 | −382.84±.00 | −1.03e4±3.58e3 | −1.03e4±3.58e3 | −1.03e4±3.58e3 | −1.03e4±3.58e3 | −382.84±.00 | −1.03e4±3.61e3 | −1.03e4±3.61e3 | −1.03e4±3.61e3 | −1.03e4±3.61e3 |
| PRIME | −379.59±1.78 | −1.14e4±195.53 | −1.14e4±195.53 | −1.14e4±195.53 | −1.14e4±195.53 | −379.59±1.78 | −1.15e4±196.54 | −1.15e4±196.54 | −1.15e4±196.54 | −1.15e4±196.54 | −379.59±1.78 | −1.15e4±196.88 | −1.15e4±196.88 | −1.15e4±196.88 | −1.15e4±196.88 |
| Task f(x*_OFF) | | | | | | CEC 5  −8.61±.14 | | | | | | | | | |
| CARCOO | −58.62±3.36 | −5.52±.49 | −6.75±.43 | −6.07±.47 | −6.28±.46 | −58.62±3.36 | −5.65±.48 | −6.91±.41 | −6.21±.45 | −6.02±.46 | −58.62±3.36 | −5.69±.47 | −6.96±.41 | −6.26±.44 | −5.70±.47 |
| DEPF | −58.84±3.83 | −9.70±1.17 | −9.70±1.17 | −9.70±1.17 | −9.70±1.17 | −62.08±4.17 | −9.94±1.02 | −9.94±1.02 | −9.94±1.02 | −9.94±1.02 | −62.88±3.87 | −10.14±.87 | −10.14±.87 | −10.14±.87 | −10.14±.87 |
| DESPF | −67.21±.20 | −12.95±3.46 | −12.95±3.46 | −12.95±3.46 | −12.95±3.46 | −67.21±.18 | −13.67±3.74 | −13.67±3.74 | −13.67±3.74 | −13.67±3.74 | −67.21±.18 | −13.91±3.84 | −13.91±3.84 | −13.91±3.84 | −13.91±3.84 |
| PRIME | −58.62±3.36 | −8.50±.71 | −8.50±.71 | −8.50±.71 | −8.50±.71 | −58.62±3.36 | −8.55±.72 | −8.55±.72 | −8.55±.72 | −8.55±.72 | −58.62±3.36 | −8.57±.72 | −8.57±.72 | −8.57±.72 | −8.57±.72 |

Table 76: Overall results for CEC constrained tasks with 128 solutions and 50th percentile evaluations. In this case, 0% of the values are missing near the worst value and another 30% near the optimal value. Details are the same as Table 5.

| Steps | t = 50 | | | | | t = 100 | | | | | t = 150 | | | | |
|---|---|---|---|---|---|---|---|---|---|---|---|---|---|---|---|
| Task f(x*_OFF) | | | | | | CEC 1  −2.08e4±424.97 | | | | | | | | | |
| Metric | FS | SI | OI | SO | SO_ω | FS | SI | OI | SO | SO_ω | FS | SI | OI | SO | SO_ω |
| CARCOO | - | - | - | - | - | - | - | - | - | - | - | - | - | - | - |
| DEPF | −2.08e4±424.97 | 1.00±.00 | 1.00±.00 | 1.00±.00 | 1.00±.00 | −2.08e4±424.97 | 1.00±.00 | 1.00±.00 | 1.00±.00 | 1.00±.00 | −2.08e4±424.97 | 1.00±.00 | 1.00±.00 | 1.00±.00 | 1.00±.00 |
| DESPF | −2.08e4±424.97 | 1.00±.00 | 1.00±.00 | 1.00±.00 | 1.00±.00 | −2.08e4±424.97 | 1.00±.00 | 1.00±.00 | 1.00±.00 | 1.00±.00 | −2.08e4±424.97 | 1.00±.00 | 1.00±.00 | 1.00±.00 | 1.00±.00 |
| PRIME | - | - | - | - | - | - | - | - | - | - | - | - | - | - | - |
| Task f(x*_OFF) | | | | | | CEC 2  −1.08±.00 | | | | | | | | | |
| CARCOO | −2.04±.00 | −93.83±1.21 | −93.83±1.21 | −93.83±1.21 | −93.83±1.21 | −2.04±.00 | −94.55±.75 | −94.55±.75 | −94.55±.75 | −94.55±.75 | −2.04±.00 | −94.79±.66 | −94.79±.66 | −94.79±.66 | −94.79±.66 |
| DEPF | −1.86±.32 | −85.72±16.08 | −85.72±16.08 | −85.72±16.08 | −85.72±16.08 | −1.95±.26 | −83.12±16.34 | −83.12±16.34 | −83.12±16.34 | −83.12±16.34 | −1.95±.26 | −83.85±15.63 | −83.85±15.63 | −83.85±15.63 | −83.85±15.63 |
| DESPF | −2.04±.00 | −94.28±.62 | −94.28±.62 | −94.28±.62 | −94.28±.62 | −2.04±.00 | −94.78±.63 | −94.78±.63 | −94.78±.63 | −94.78±.63 | −2.04±.00 | −94.94±.63 | −94.94±.63 | −94.94±.63 | −94.94±.63 |
| PRIME | −2.04±.00 | −94.28±.62 | −94.28±.62 | −94.28±.62 | −94.28±.62 | −2.04±.00 | −94.78±.63 | −94.78±.63 | −94.78±.63 | −94.78±.63 | −2.04±.00 | −94.94±.63 | −94.94±.63 | −94.94±.63 | −94.94±.63 |
| Task f(x*_OFF) | | | | | | CEC 3  −2.0±.00 | | | | | | | | | |
| CARCOO | −4.20±.00 | −216.35±.20 | −216.35±.20 | −216.35±.20 | −216.35±.20 | −4.20±.00 | −217.48±.20 | −217.48±.20 | −217.48±.20 | −217.48±.20 | −4.20±.00 | −217.85±.20 | −217.85±.20 | −217.85±.20 | −217.85±.20 |
| DEPF | −4.20±.00 | −216.35±.20 | −216.35±.20 | −216.35±.20 | −216.35±.20 | −4.20±.00 | −217.48±.20 | −217.48±.20 | −217.48±.20 | −217.48±.20 | −4.20±.00 | −217.85±.20 | −217.85±.20 | −217.85±.20 | −217.85±.20 |
| DESPF | −4.20±.00 | −216.35±.20 | −216.35±.20 | −216.35±.20 | −216.35±.20 | −4.20±.00 | −217.48±.20 | −217.48±.20 | −217.48±.20 | −217.48±.20 | −4.20±.00 | −217.85±.20 | −217.85±.20 | −217.85±.20 | −217.85±.20 |
| PRIME | −4.20±.00 | −216.35±.20 | −216.35±.20 | −216.35±.20 | −216.35±.20 | −4.20±.00 | −217.48±.20 | −217.48±.20 | −217.48±.20 | −217.48±.20 | −4.20±.00 | −217.85±.20 | −217.85±.20 | −217.85±.20 | −217.85±.20 |
| Task f(x*_OFF) | | | | | | CEC 4  −264.43±.30 | | | | | | | | | |
| CARCOO | −379.59±1.78 | −1.14e4±195.53 | −1.14e4±195.53 | −1.14e4±195.53 | −1.14e4±195.53 | −379.59±1.78 | −1.15e4±196.54 | −1.15e4±196.54 | −1.15e4±196.54 | −1.15e4±196.54 | −379.59±1.78 | −1.15e4±196.88 | −1.15e4±196.88 | −1.15e4±196.88 | −1.15e4±196.88 |
| DEPF | −382.84±.00 | −1.02e4±3.58e3 | −1.02e4±3.58e3 | −1.02e4±3.58e3 | −1.02e4±3.58e3 | −382.84±.00 | −1.03e4±3.63e3 | −1.03e4±3.63e3 | −1.03e4±3.63e3 | −1.03e4±3.63e3 | −382.84±.00 | −1.04e4±3.64e3 | −1.04e4±3.64e3 | −1.04e4±3.64e3 | −1.04e4±3.64e3 |
| DESPF | −382.42±.37 | −7.41e3±3.14e3 | −7.41e3±3.14e3 | −7.41e3±3.14e3 | −7.41e3±3.14e3 | −382.82±.02 | −7.58e3±3.24e3 | −7.58e3±3.24e3 | −7.58e3±3.24e3 | −7.58e3±3.24e3 | −382.83±.02 | −7.63e3±3.27e3 | −7.63e3±3.27e3 | −7.63e3±3.27e3 | −7.63e3±3.27e3 |
| PRIME | −379.59±1.78 | −1.01e4±3.52e3 | −1.01e4±3.52e3 | −1.01e4±3.52e3 | −1.01e4±3.52e3 | −379.59±1.78 | −1.01e4±3.54e3 | −1.01e4±3.54e3 | −1.01e4±3.54e3 | −1.01e4±3.54e3 | −379.59±1.78 | −1.01e4±3.55e3 | −1.01e4±3.55e3 | −1.01e4±3.55e3 | −1.01e4±3.55e3 |
| Task f(x*_OFF) | | | | | | CEC 5  −6.75±.04 | | | | | | | | | |
| CARCOO | −58.62±3.36 | −9.24±.48 | −10.58±.63 | −9.86±.52 | −10.09±.54 | −58.62±3.36 | −9.40±.49 | −10.76±.04 | −10.04±.53 | −9.82±.51 | −58.62±3.36 | −9.46±.49 | −10.82±.64 | −10.09±.53 | −9.47±.49 |
| DEPF | −58.62±3.36 | −16.71±6.59 | −16.71±6.59 | −16.71±6.59 | −16.71±6.59 | −61.75±2.04 | −16.98±6.54 | −16.98±6.54 | −16.98±6.54 | −16.98±6.54 | −62.73±3.29 | −17.33±6.43 | −17.33±6.43 | −17.33±6.43 | −17.33±6.43 |
| DESPF | −67.28±.00 | −103.16±211.41 | −103.16±211.41 | −103.16±211.41 | −103.16±211.41 | −67.28±.00 | −107.80±219.72 | −107.80±219.72 | −107.80±219.72 | −107.80±219.72 | −67.28±.00 | −109.33±222.46 | −109.33±222.46 | −109.33±222.46 | −109.33±222.46 |
| PRIME | −58.62±3.36 | −12.40±1.38 | −12.40±1.38 | −12.40±1.38 | −12.40±1.38 | −58.62±3.36 | −12.47±1.39 | −12.47±1.39 | −12.47±1.39 | −12.47±1.39 | −58.62±3.36 | −12.49±1.39 | −12.49±1.39 | −12.49±1.39 | −12.49±1.39 |

Table 77: Overall results for CEC constrained tasks with 128 solutions and 50th percentile evaluations. In this case, 0% of the values are missing near the worst value and another 20% near the optimal value. Details are the same as Table 5.

| Steps | t = 50 | | | | | t = 100 | | | | | t = 150 | | | | |
|---|---|---|---|---|---|---|---|---|---|---|---|---|---|---|---|
| Task f(x*_OFF) | | | | | | CEC 1  −2.08e4±424.97 | | | | | | | | | |
| Metric | FS | SI | OI | SO | SO_ω | FS | SI | OI | SO | SO_ω | FS | SI | OI | SO | SO_ω |
| CARCOO | - | - | - | - | - | - | - | - | - | - | - | - | - | - | - |
| DEPF | −2.08e4±424.97 | 1.00±.00 | 1.00±.00 | 1.00±.00 | 1.00±.00 | −2.08e4±424.97 | 1.00±.00 | 1.00±.00 | 1.00±.00 | 1.00±.00 | −2.08e4±424.97 | 1.00±.00 | 1.00±.00 | 1.00±.00 | 1.00±.00 |
| DESPF | −2.08e4±424.97 | 1.00±.00 | 1.00±.00 | 1.00±.00 | 1.00±.00 | −2.08e4±424.97 | 1.00±.00 | 1.00±.00 | 1.00±.00 | 1.00±.00 | −2.08e4±424.97 | 1.00±.00 | 1.00±.00 | 1.00±.00 | 1.00±.00 |
| PRIME | - | - | - | - | - | - | - | - | - | - | - | - | - | - | - |
| Task f(x*_OFF) | | | | | | CEC 2  −1.08±.00 | | | | | | | | | |
| CARCOO | −2.04±.00 | −94.08±.56 | −94.08±.56 | −94.08±.56 | −94.08±.56 | −2.04±.00 | −94.68±.53 | −94.68±.53 | −94.68±.53 | −94.68±.53 | −2.04±.00 | −94.87±.55 | −94.87±.55 | −94.87±.55 | −94.87±.55 |
| DEPF | −1.87±.40 | −87.21±12.59 | −87.21±12.59 | −87.21±12.59 | −87.21±12.59 | −2.00±.12 | −83.95±19.03 | −83.95±19.03 | −83.95±19.03 | −83.95±19.03 | −1.96±.23 | −85.53±17.38 | −85.53±17.38 | −85.53±17.38 | −85.53±17.38 |
| DESPF | −2.04±.00 | −94.28±.62 | −94.28±.62 | −94.28±.62 | −94.28±.62 | −2.04±.00 | −94.78±.63 | −94.78±.63 | −94.78±.63 | −94.78±.63 | −2.04±.00 | −94.94±.63 | −94.94±.63 | −94.94±.63 | −94.94±.63 |
| PRIME | −2.04±.00 | −94.28±.62 | −94.28±.62 | −94.28±.62 | −94.28±.62 | −2.04±.00 | −94.78±.63 | −94.78±.63 | −94.78±.63 | −94.78±.63 | −2.04±.00 | −94.94±.63 | −94.94±.63 | −94.94±.63 | −94.94±.63 |
| Task f(x*_OFF) | | | | | | CEC 3  −2.0±.00 | | | | | | | | | |
| CARCOO | −2.87±1.03 | −66.99±87.36 | −66.99±87.36 | −66.99±87.36 | −66.99±87.36 | −2.87±1.03 | −74.41±90.62 | −74.41±90.62 | −74.41±90.62 | −74.41±90.62 | −2.88±1.02 | −77.63±93.08 | −77.63±93.08 | −77.63±93.08 | −77.63±93.08 |
| DEPF | −4.20±.00 | −216.35±.20 | −216.35±.20 | −216.35±.20 | −216.35±.20 | −4.20±.00 | −217.48±.20 | −217.48±.20 | −217.48±.20 | −217.48±.20 | −4.20±.00 | −217.85±.20 | −217.85±.20 | −217.85±.20 | −217.85±.20 |
| DESPF | −4.20±.00 | −216.35±.20 | −216.35±.20 | −216.35±.20 | −216.35±.20 | −4.20±.00 | −217.48±.20 | −217.48±.20 | −217.48±.20 | −217.48±.20 | −4.20±.00 | −217.85±.20 | −217.85±.20 | −217.85±.20 | −217.85±.20 |
| PRIME | −4.20±.00 | −216.35±.20 | −216.35±.20 | −216.35±.20 | −216.35±.20 | −4.20±.00 | −217.48±.20 | −217.48±.20 | −217.48±.20 | −217.48±.20 | −4.20±.00 | −217.85±.20 | −217.85±.20 | −217.85±.20 | −217.85±.20 |
| Task f(x*_OFF) | | | | | | CEC 4  −264.43±.30 | | | | | | | | | |
| CARCOO | −379.59±1.78 | −1.14e4±195.53 | −1.14e4±195.53 | −1.14e4±195.53 | −1.14e4±195.53 | −379.59±1.78 | −1.15e4±196.54 | −1.15e4±196.54 | −1.15e4±196.54 | −1.15e4±196.54 | −379.59±1.78 | −1.15e4±196.88 | −1.15e4±196.88 | −1.15e4±196.88 | −1.15e4±196.88 |
| DEPF | −382.84±.00 | −1.02e4±3.42e3 | −1.02e4±3.42e3 | −1.02e4±3.42e3 | −1.02e4±3.42e3 | −382.84±.00 | −1.04e4±3.47e3 | −1.04e4±3.47e3 | −1.04e4±3.47e3 | −1.04e4±3.47e3 | −382.84±.00 | −1.04e4±3.48e3 | −1.04e4±3.48e3 | −1.04e4±3.48e3 | −1.04e4±3.48e3 |
| DESPF | −382.54±.31 | −9.87e3±3.60e3 | −9.87e3±3.60e3 | −9.87e3±3.60e3 | −9.87e3±3.60e3 | −382.83±.01 | −1.01e4±3.77e3 | −1.01e4±3.77e3 | −1.01e4±3.77e3 | −1.01e4±3.77e3 | −382.84±.00 | −1.02e4±3.81e3 | −1.02e4±3.81e3 | −1.02e4±3.81e3 | −1.02e4±3.81e3 |
| PRIME | −379.59±1.78 | −9.43e3±3.73e3 | −9.43e3±3.73e3 | −9.43e3±3.73e3 | −9.43e3±3.73e3 | −379.59±1.78 | −9.48e3±3.74e3 | −9.48e3±3.74e3 | −9.48e3±3.74e3 | −9.48e3±3.74e3 | −379.59±1.78 | −9.49e3±3.75e3 | −9.49e3±3.75e3 | −9.49e3±3.75e3 | −9.49e3±3.75e3 |
| Task f(x*_OFF) | | | | | | CEC 5  −4.91±.10 | | | | | | | | | |
| CARCOO | −58.59±3.59 | −18.55±1.86 | −18.55±1.86 | −18.55±1.86 | −18.55±1.86 | −58.59±3.59 | −18.73±1.94 | −18.73±1.94 | −18.73±1.94 | −18.73±1.94 | −58.59±3.59 | −18.79±1.97 | −18.79±1.97 | −18.79±1.97 | −18.79±1.97 |
| DEPF | −59.33±4.37 | −669.00±1.64e3 | −669.00±1.64e3 | −669.00±1.64e3 | −669.00±1.64e3 | −62.29±2.52 | −682.70±1.68e3 | −682.70±1.68e3 | −682.70±1.68e3 | −682.70±1.68e3 | −64.39±2.79 | −715.09±1.76e3 | −715.09±1.76e3 | −715.09±1.76e3 | −715.09±1.76e3 |
| DESPF | −67.26±.04 | −2.08e3±2.62e3 | −2.08e3±2.62e3 | −2.08e3±2.62e3 | −2.08e3±2.62e3 | −67.28±.00 | −2.23e3±2.81e3 | −2.23e3±2.81e3 | −2.23e3±2.81e3 | −2.23e3±2.81e3 | −67.28±.00 | −2.28e3±2.87e3 | −2.28e3±2.87e3 | −2.28e3±2.87e3 | −2.28e3±2.87e3 |
| PRIME | −58.62±3.36 | −26.59±4.04 | −26.59±4.04 | −26.59±4.04 | −26.59±4.04 | −58.62±3.36 | −26.73±4.06 | −26.73±4.06 | −26.73±4.06 | −26.73±4.06 | −58.62±3.36 | −26.78±4.07 | −26.78±4.07 | −26.78±4.07 | −26.78±4.07 |

Table 78: Overall results for CEC constrained tasks with 128 solutions and 50th percentile evaluations. In this case, 0% of the values are missing near the worst value and another 10% near the optimal value. Details are the same as Table 5.

| Steps | t = 50 | | | | | t = 100 | | | | | t = 150 | | | | |
|---|---|---|---|---|---|---|---|---|---|---|---|---|---|---|---|
| **Task** $f(x^*_{OFF})$ | CEC 1 −2.08e4±424.97 | | | | | | | | | | | | | | |
| Metric | FS | SI | OI | SO | SO_ω | FS | SI | OI | SO | SO_ω | FS | SI | OI | SO | SO_ω |
| CARCOO | - | - | - | - | - | - | - | - | - | - | - | - | - | - | - |
| DEPF | −2.08e4±424.97 | 1.00±.00 | 1.00±.00 | 1.00±.00 | 1.00±.00 | −2.08e4±424.97 | 1.00±.00 | 1.00±.00 | 1.00±.00 | 1.00±.00 | −2.08e4±424.97 | 1.00±.00 | 1.00±.00 | 1.00±.00 | 1.00±.00 |
| DESPF | −2.08e4±424.97 | 1.00±.00 | 1.00±.00 | 1.00±.00 | 1.00±.00 | −2.08e4±424.97 | 1.00±.00 | 1.00±.00 | 1.00±.00 | 1.00±.00 | −2.08e4±424.97 | 1.00±.00 | 1.00±.00 | 1.00±.00 | 1.00±.00 |
| PRIME | - | - | - | - | - | - | - | - | - | - | - | - | - | - | - |
| **Task** $f(x^*_{OFF})$ | CEC 2 −1.08±0.00 | | | | | | | | | | | | | | |
| CARCOO | −2.04±.00 | −93.90±1.45 | −93.90±1.45 | −93.90±1.45 | −93.90±1.45 | −2.04±.00 | −94.59±1.00 | −94.59±1.00 | −94.59±1.00 | −94.59±1.00 | −2.04±.00 | −94.81±.86 | −94.81±.86 | −94.81±.86 | −94.81±.86 |
| DEPF | −1.96±.23 | −89.93±11.45 | −89.93±11.45 | −89.93±11.45 | −89.93±11.45 | −1.87±.30 | −84.99±18.30 | −84.99±18.30 | −84.99±18.30 | −84.99±18.30 | −1.96±.23 | −84.59±19.86 | −84.59±19.86 | −84.59±19.86 | −84.59±19.86 |
| DESPF | −2.04±.00 | −94.28±.62 | −94.28±.62 | −94.28±.62 | −94.28±.62 | −2.04±.00 | −94.78±.63 | −94.78±.63 | −94.78±.63 | −94.78±.63 | −2.04±.00 | −94.94±.63 | −94.94±.63 | −94.94±.63 | −94.94±.63 |
| PRIME | −2.04±.00 | −94.28±.62 | −94.28±.62 | −94.28±.62 | −94.28±.62 | −2.04±.00 | −94.78±.63 | −94.78±.63 | −94.78±.63 | −94.78±.63 | −2.04±.00 | −94.94±.63 | −94.94±.63 | −94.94±.63 | −94.94±.63 |
| **Task** $f(x^*_{OFF})$ | CEC 3 −2.0±0.0 | | | | | | | | | | | | | | |
| CARCOO | −2.10±.02 | −8.69±1.97 | −8.69±1.97 | −8.69±1.97 | −8.69±1.97 | −2.40±.73 | −18.04±23.17 | −18.04±23.17 | −18.04±23.17 | −18.04±23.17 | −2.40±.73 | −25.00±40.06 | −25.00±40.06 | −25.00±40.06 | −25.00±40.06 |
| DEPF | −4.20±.00 | −216.35±.20 | −216.35±.20 | −216.35±.20 | −216.35±.20 | −4.20±.00 | −217.48±.20 | −217.48±.20 | −217.48±.20 | −217.48±.20 | −4.20±.00 | −217.85±.20 | −217.85±.20 | −217.85±.20 | −217.85±.20 |
| DESPF | −4.20±.00 | −216.35±.20 | −216.35±.20 | −216.35±.20 | −216.35±.20 | −4.20±.00 | −217.48±.20 | −217.48±.20 | −217.48±.20 | −217.48±.20 | −4.20±.00 | −217.85±.20 | −217.85±.20 | −217.85±.20 | −217.85±.20 |
| PRIME | −4.20±.00 | −216.35±.20 | −216.35±.20 | −216.35±.20 | −216.35±.20 | −4.20±.00 | −217.48±.20 | −217.48±.20 | −217.48±.20 | −217.48±.20 | −4.20±.00 | −217.85±.20 | −217.85±.20 | −217.85±.20 | −217.85±.20 |
| **Task** $f(x^*_{OFF})$ | CEC 4 −264.43±0.00 | | | | | | | | | | | | | | |
| CARCOO | −379.59±1.78 | −1.14e4±195.53 | −1.14e4±195.53 | −1.14e4±195.53 | −1.14e4±195.53 | −379.59±1.78 | −1.15e4±196.54 | −1.15e4±196.54 | −1.15e4±196.54 | −1.15e4±196.54 | −379.59±1.78 | −1.15e4±196.88 | −1.15e4±196.88 | −1.15e4±196.88 | −1.15e4±196.88 |
| DEPF | −381.37±2.06 | −8.75e3±4.78e3 | −8.75e3±4.78e3 | −8.75e3±4.78e3 | −8.75e3±4.78e3 | −382.64±0.53 | −8.87e3±4.84e3 | −8.87e3±4.84e3 | −8.87e3±4.84e3 | −8.87e3±4.84e3 | −382.84±0.00 | −8.92e3±4.86e3 | −8.92e3±4.86e3 | −8.92e3±4.86e3 | −8.92e3±4.86e3 |
| DESPF | −382.24±0.25 | −1.10e4±232.23 | −1.10e4±232.23 | −1.10e4±232.23 | −1.10e4±232.23 | −382.79±0.04 | −1.14e4±123.73 | −1.14e4±123.73 | −1.14e4±123.73 | −1.14e4±123.73 | −382.80±0.04 | −1.16e4±87.59 | −1.16e4±87.59 | −1.16e4±87.59 | −1.16e4±87.59 |
| PRIME | −379.59±1.78 | −1.01e4±3.68e3 | −1.01e4±3.68e3 | −1.01e4±3.68e3 | −1.01e4±3.68e3 | −379.59±1.78 | −1.01e4±3.70e3 | −1.01e4±3.70e3 | −1.01e4±3.70e3 | −1.01e4±3.70e3 | −379.59±1.78 | −1.01e4±3.71e3 | −1.01e4±3.71e3 | −1.01e4±3.71e3 | −1.01e4±3.71e3 |
| **Task** $f(x^*_{OFF})$ | CEC 5 −2.95±0.06 | | | | | | | | | | | | | | |
| CARCOO | −58.62±3.36 | −88.00±30.42 | −88.00±30.42 | −88.00±30.42 | −88.00±30.42 | −58.62±3.36 | −88.46±30.58 | −88.46±30.58 | −88.46±30.58 | −88.46±30.58 | −58.62±3.36 | −88.62±30.63 | −88.62±30.63 | −88.62±30.63 | −88.62±30.63 |
| DEPF | −58.09±2.44 | −4.90e3±1.61e3 | −4.90e3±1.61e3 | −4.90e3±1.61e3 | −4.90e3±1.61e3 | −59.16±1.01 | −4.92e3±1.61e3 | −4.92e3±1.61e3 | −4.92e3±1.61e3 | −4.92e3±1.61e3 | −60.83±4.09 | −4.96e3±1.62e3 | −4.96e3±1.62e3 | −4.96e3±1.62e3 | −4.96e3±1.62e3 |
| DESPF | −65.75±3.87 | −5.69e3±300.25 | −5.69e3±300.25 | −5.69e3±300.25 | −5.69e3±300.25 | −65.81±1.89 | −5.99e3±308.29 | −5.99e3±308.29 | −5.99e3±308.29 | −5.99e3±308.29 | −65.81±1.89 | −6.09e3±328.20 | −6.09e3±328.20 | −6.09e3±328.20 | −6.09e3±328.20 |
| PRIME | −58.62±3.36 | −4.83e3±1.83e3 | −4.83e3±1.83e3 | −4.83e3±1.83e3 | −4.83e3±1.83e3 | −58.62±3.36 | −4.85e3±1.84e3 | −4.85e3±1.84e3 | −4.85e3±1.84e3 | −4.85e3±1.84e3 | −58.62±3.36 | −4.86e3±1.85e3 | −4.86e3±1.85e3 | −4.86e3±1.85e3 | −4.86e3±1.85e3 |

Table 79: Overall results for CEC constrained tasks with 128 solutions and 50th percentile evaluations. In this case, 10% of the values are missing near the worst value and another 40% near the optimal value. Details are the same as Table 5.

| Steps | t = 50 | | | | | t = 100 | | | | | t = 150 | | | | |
|---|---|---|---|---|---|---|---|---|---|---|---|---|---|---|---|
| **Task** $f(x^*_{OFF})$ | CEC 1 −1.39e4±94.68 | | | | | | | | | | | | | | |
| Metric | FS | SI | OI | SO | SO_ω | FS | SI | OI | SO | SO_ω | FS | SI | OI | SO | SO_ω |
| CARCOO | - | - | - | - | - | - | - | - | - | - | - | - | - | - | - |
| DEPF | −1.39e4±94.68 | 1.00±.00 | 1.00±.00 | 1.00±.00 | 1.00±.00 | −1.39e4±94.68 | 1.00±.00 | 1.00±.00 | 1.00±.00 | 1.00±.00 | −1.39e4±94.68 | 1.00±.00 | 1.00±.00 | 1.00±.00 | 1.00±.00 |
| DESPF | −1.39e4±94.68 | 1.00±.00 | 1.00±.00 | 1.00±.00 | 1.00±.00 | −1.39e4±94.68 | 1.00±.00 | 1.00±.00 | 1.00±.00 | 1.00±.00 | −1.39e4±94.68 | 1.00±.00 | 1.00±.00 | 1.00±.00 | 1.00±.00 |
| PRIME | - | - | - | - | - | - | - | - | - | - | - | - | - | - | - |
| **Task** $f(x^*_{OFF})$ | CEC 2 −1.08±0.00 | | | | | | | | | | | | | | |
| CARCOO | −1.33±.03 | −23.10±2.46 | −23.10±2.46 | −23.10±2.46 | −23.10±2.46 | −1.33±.03 | −23.31±2.50 | −23.31±2.50 | −23.31±2.50 | −23.31±2.50 | −1.33±.03 | −23.38±2.52 | −23.38±2.52 | −23.38±2.52 | −23.38±2.52 |
| DEPF | −1.33±.03 | −24.05±.97 | −24.05±.97 | −24.05±.97 | −24.05±.97 | −1.33±.03 | −23.73±1.71 | −23.73±1.71 | −23.73±1.71 | −23.73±1.71 | −1.33±.03 | −23.63±2.10 | −23.63±2.10 | −23.63±2.10 | −23.63±2.10 |
| DESPF | −1.34±.00 | −24.27±.33 | −24.27±.33 | −24.27±.33 | −24.27±.33 | −1.34±.00 | −24.40±.33 | −24.40±.33 | −24.40±.33 | −24.40±.33 | −1.34±.00 | −24.45±.33 | −24.45±.33 | −24.45±.33 | −24.45±.33 |
| PRIME | −1.34±.00 | −24.27±.33 | −24.27±.33 | −24.27±.33 | −24.27±.33 | −1.34±.00 | −24.40±.33 | −24.40±.33 | −24.40±.33 | −24.40±.33 | −1.34±.00 | −24.45±.33 | −24.45±.33 | −24.45±.33 | −24.45±.33 |
| **Task** $f(x^*_{OFF})$ | CEC 3 −2.12±0.02 | | | | | | | | | | | | | | |
| CARCOO | −3.36±.05 | −24.71±38.56 | −24.71±38.56 | −24.71±38.56 | −24.71±38.56 | −3.36±.05 | −24.86±38.76 | −24.86±38.76 | −24.86±38.76 | −24.86±38.76 | −3.36±.05 | −24.91±38.82 | −24.91±38.82 | −24.91±38.82 | −24.91±38.82 |
| DEPF | −3.35±.07 | −10.68±1.94 | −10.68±1.94 | −10.68±1.94 | −10.68±1.94 | −3.35±.07 | −10.72±1.96 | −10.72±1.96 | −10.72±1.96 | −10.72±1.96 | −3.35±.07 | −10.74±1.97 | −10.74±1.97 | −10.74±1.97 | −10.74±1.97 |
| DESPF | −3.34±.08 | −24.24±32.56 | −24.24±32.56 | −24.24±32.56 | −24.24±32.56 | −3.34±.08 | −24.21±32.34 | −24.21±32.34 | −24.21±32.34 | −24.21±32.34 | −3.34±.08 | −24.20±32.26 | −24.20±32.26 | −24.20±32.26 | −24.20±32.26 |
| PRIME | −3.36±.05 | −9.48±1.22 | −9.48±1.22 | −9.48±1.22 | −9.48±1.22 | −3.36±.05 | −9.55±1.25 | −9.55±1.25 | −9.55±1.25 | −9.55±1.25 | −3.36±.05 | −9.58±1.26 | −9.58±1.26 | −9.58±1.26 | −9.58±1.26 |
| **Task** $f(x^*_{OFF})$ | CEC 4 −264.43±0.30 | | | | | | | | | | | | | | |
| CARCOO | −327.69±1.16 | −6.26e3±110.35 | −6.26e3±110.35 | −6.26e3±110.35 | −6.26e3±110.35 | −327.69±1.16 | −6.29e3±110.92 | −6.29e3±110.92 | −6.29e3±110.92 | −6.29e3±110.92 | −327.69±1.16 | −6.30e3±111.11 | −6.30e3±111.11 | −6.30e3±111.11 | −6.30e3±111.11 |
| DEPF | −382.84±0.00 | −1.04e4±57.29 | −1.04e4±57.29 | −1.04e4±57.29 | −1.04e4±57.29 | −382.84±0.00 | −1.11e4±35.72 | −1.11e4±35.72 | −1.11e4±35.72 | −1.11e4±35.72 | −382.84±0.00 | −1.14e4±31.25 | −1.14e4±31.25 | −1.14e4±31.25 | −1.14e4±31.25 |
| DESPF | −382.73±0.30 | −9.23e3±2.91e3 | −9.23e3±2.91e3 | −9.23e3±2.91e3 | −9.23e3±2.91e3 | −382.84±0.01 | −9.91e3±3.11e3 | −9.91e3±3.11e3 | −9.91e3±3.11e3 | −9.91e3±3.11e3 | −382.84±0.00 | −1.01e4±3.18e3 | −1.01e4±3.18e3 | −1.01e4±3.18e3 | −1.01e4±3.18e3 |
| PRIME | −327.69±1.16 | −6.26e3±110.35 | −6.26e3±110.35 | −6.26e3±110.35 | −6.26e3±110.35 | −327.69±1.16 | −6.29e3±110.92 | −6.29e3±110.92 | −6.29e3±110.92 | −6.29e3±110.92 | −327.69±1.16 | −6.30e3±111.11 | −6.30e3±111.11 | −6.30e3±111.11 | −6.30e3±111.11 |
| **Task** $f(x^*_{OFF})$ | CEC 5 −8.61±1.61 | | | | | | | | | | | | | | |
| CARCOO | −27.28±.15 | −1.61±.10 | −1.94±.10 | −1.76±.10 | −1.82±.10 | −27.28±.15 | −1.66±.10 | −1.99±.10 | −1.81±.10 | −1.75±.10 | −27.28±.15 | −1.67±.10 | −2.00±.10 | −1.82±.10 | −1.67±.08 |
| DEPF | −27.27±.14 | −3.05±.96 | −3.05±.96 | −3.05±.96 | −3.05±.96 | −50.73±18.24 | −4.14±.81 | −4.14±.81 | −4.14±.81 | −4.14±.81 | −53.15±16.33 | −5.43±1.68 | −5.43±1.68 | −5.43±1.65 | −5.43±1.65 |
| DESPF | −67.28±.00 | −10.56±3.62 | −10.56±3.62 | −10.56±3.62 | −10.56±3.62 | −67.28±.00 | −12.87±4.40 | −12.87±4.40 | −12.87±4.40 | −12.87±4.40 | −67.28±.00 | −13.62±4.67 | −13.62±4.67 | −13.62±4.67 | −13.62±4.67 |
| PRIME | −27.27±.14 | −2.40±.27 | −2.40±.27 | −2.40±.27 | −2.40±.27 | −27.27±.14 | −2.42±.27 | −2.42±.27 | −2.42±.27 | −2.42±.27 | −27.27±.14 | −2.42±.27 | −2.42±.27 | −2.42±.27 | −2.42±.27 |

Table 80: Overall results for CEC constrained tasks with 128 solutions and 50th percentile evaluations. In this case, 20% of the values are missing near the worst value and another 30% near the optimal value. Details are the same as Table 5.

| Steps | t = 50 | | | | | t = 100 | | | | | t = 150 | | | | |
|---|---|---|---|---|---|---|---|---|---|---|---|---|---|---|---|
| **Task** $f(x^*_{OFF})$ | CEC 1 −1.17e4±74.77 | | | | | | | | | | | | | | |
| Metric | FS | SI | OI | SO | SO_ω | FS | SI | OI | SO | SO_ω | FS | SI | OI | SO | SO_ω |
| CARCOO | - | - | - | - | - | - | - | - | - | - | - | - | - | - | - |
| DEPF | −1.17e4±74.77 | 1.00±.00 | 1.00±.00 | 1.00±.00 | 1.00±.00 | −1.17e4±74.77 | 1.00±.00 | 1.00±.00 | 1.00±.00 | 1.00±.00 | −1.17e4±74.77 | 1.00±.00 | 1.00±.00 | 1.00±.00 | 1.00±.00 |
| DESPF | −1.17e4±74.77 | 1.00±.00 | 1.00±.00 | 1.00±.00 | 1.00±.00 | −1.17e4±74.77 | 1.00±.00 | 1.00±.00 | 1.00±.00 | 1.00±.00 | −1.17e4±74.77 | 1.00±.00 | 1.00±.00 | 1.00±.00 | 1.00±.00 |
| PRIME | - | - | - | - | - | - | - | - | - | - | - | - | - | - | - |
| **Task** $f(x^*_{OFF})$ | CEC 2 −1.08±0.00 | | | | | | | | | | | | | | |
| CARCOO | −1.30±.00 | −20.69±.18 | −20.69±.18 | −20.69±.18 | −20.69±.18 | −1.30±.00 | −20.84±.19 | −20.84±.19 | −20.84±.19 | −20.84±.19 | −1.30±.00 | −20.88±.20 | −20.88±.20 | −20.88±.20 | −20.88±.20 |
| DEPF | −1.30±.00 | −20.64±.20 | −20.64±.20 | −20.64±.20 | −20.64±.20 | −1.29±.02 | −20.22±.98 | −20.22±.98 | −20.22±.98 | −20.22±.98 | −1.29±.02 | −20.20±1.22 | −20.20±1.22 | −20.20±1.22 | −20.20±1.22 |
| DESPF | −1.30±.00 | −20.73±.20 | −20.73±.20 | −20.73±.20 | −20.73±.20 | −1.30±.00 | −20.84±.20 | −20.84±.20 | −20.84±.20 | −20.84±.20 | −1.30±.00 | −20.88±.20 | −20.88±.20 | −20.88±.20 | −20.88±.20 |
| PRIME | −1.30±.00 | −20.73±.20 | −20.73±.20 | −20.73±.20 | −20.73±.20 | −1.30±.00 | −20.84±.20 | −20.84±.20 | −20.84±.20 | −20.84±.20 | −1.30±.00 | −20.88±.20 | −20.88±.20 | −20.88±.20 | −20.88±.20 |
| **Task** $f(x^*_{OFF})$ | CEC 3 −2.0±0.0 | | | | | | | | | | | | | | |
| CARCOO | −2.90±.32 | −86.66±31.18 | −86.66±31.18 | −86.66±31.18 | −86.66±31.18 | −2.90±.32 | −87.78±31.59 | −87.78±31.59 | −87.78±31.59 | −87.78±31.59 | −2.90±.32 | −88.15±31.72 | −88.15±31.72 | −88.15±31.72 | −88.15±31.72 |
| DEPF | −3.16±.07 | −106.20±4.76 | −106.20±4.76 | −106.20±4.76 | −106.20±4.76 | −3.16±.07 | −110.42±5.64 | −110.42±5.64 | −110.42±5.64 | −110.42±5.64 | −3.16±.07 | −111.81±6.18 | −111.81±6.18 | −111.81±6.18 | −111.81±6.18 |
| DESPF | −3.13±.09 | −106.22±5.64 | −106.22±5.64 | −106.22±5.64 | −106.22±5.64 | −3.13±.09 | −109.18±7.06 | −109.18±7.06 | −109.18±7.06 | −109.18±7.06 | −3.13±.09 | −110.16±7.62 | −110.16±7.62 | −110.16±7.62 | −110.16±7.62 |
| PRIME | −3.02±.01 | −99.77±1.18 | −99.77±1.18 | −99.77±1.18 | −99.77±1.18 | −3.02±.01 | −100.29±1.18 | −100.29±1.18 | −100.29±1.18 | −100.29±1.18 | −3.02±.01 | −100.47±1.18 | −100.47±1.18 | −100.47±1.18 | −100.47±1.18 |
| **Task** $f(x^*_{OFF})$ | CEC 4 −264.43±0.00 | | | | | | | | | | | | | | |
| CARCOO | −304.56±.74 | −3.97e3±82.73 | −3.97e3±82.73 | −3.97e3±82.73 | −3.97e3±82.73 | −304.56±.74 | −3.99e3±83.16 | −3.99e3±83.16 | −3.99e3±83.16 | −3.99e3±83.16 | −304.56±.74 | −4.00e3±83.30 | −4.00e3±83.30 | −4.00e3±83.30 | −4.00e3±83.30 |
| DEPF | −382.84±0.00 | −1.02e4±115.32 | −1.02e4±115.32 | −1.02e4±115.32 | −1.02e4±115.32 | −382.84±0.00 | −1.10e4±59.74 | −1.10e4±59.74 | −1.10e4±59.74 | −1.10e4±59.74 | −382.84±0.00 | −1.13e4±43.80 | −1.13e4±43.80 | −1.13e4±43.80 | −1.13e4±43.80 |
| DESPF | −382.65±0.37 | −9.56e3±982.71 | −9.56e3±982.71 | −9.56e3±982.71 | −9.56e3±982.71 | −382.83±0.03 | −1.05e4±1.05e3 | −1.05e4±1.05e3 | −1.05e4±1.05e3 | −1.05e4±1.05e3 | −382.83±0.02 | −1.08e4±1.07e3 | −1.08e4±1.07e3 | −1.08e4±1.07e3 | −1.08e4±1.07e3 |
| PRIME | −304.56±.74 | −3.97e3±82.73 | −3.97e3±82.73 | −3.97e3±82.73 | −3.97e3±82.73 | −304.56±.74 | −3.99e3±83.16 | −3.99e3±83.16 | −3.99e3±83.16 | −3.99e3±83.16 | −304.56±.74 | −4.00e3±83.30 | −4.00e3±83.30 | −4.00e3±83.30 | −4.00e3±83.30 |
| **Task** $f(x^*_{OFF})$ | CEC 5 −6.75±0.10 | | | | | | | | | | | | | | |
| CARCOO | - | - | - | - | - | - | - | - | - | - | - | - | - | - | - |
| DEPF | −20.33±.24 | −3.79±1.20 | −3.79±1.20 | −3.79±1.20 | −3.79±1.20 | −33.73±15.37 | −4.60±1.28 | −4.60±1.28 | −4.60±1.28 | −4.60±1.28 | −50.43±19.71 | −7.13±2.97 | −7.13±2.97 | −7.13±2.97 | −7.13±2.97 |
| DESPF | −64.39±7.65 | −35.76±48.58 | −35.76±48.58 | −35.76±48.58 | −35.76±48.58 | −67.28±.00 | −46.06±60.17 | −46.06±60.17 | −46.06±60.17 | −46.06±60.17 | −67.28±.00 | −49.55±63.94 | −49.55±63.94 | −49.55±63.94 | −49.55±63.94 |
| PRIME | −20.31±.22 | −2.51±.30 | −2.51±.30 | −2.51±.30 | −2.51±.30 | −20.31±.22 | −2.53±.30 | −2.53±.30 | −2.53±.30 | −2.53±.30 | −20.31±.22 | −2.53±.30 | −2.53±.30 | −2.53±.30 | −2.53±.30 |

Table 81: Overall results for CEC constrained tasks with 128 solutions and 50th percentile evaluations. In this case, 25% of the values are missing near the worst value and another 25% near the optimal value. Details are the same as Table 5.

| Steps | t = 50 | | | | | t = 100 | | | | | t = 150 | | | | |
|---|---|---|---|---|---|---|---|---|---|---|---|---|---|---|---|
| Metric | FS | SI | OI | SO | $SO_\omega$ | FS | SI | OI | SO | $SO_\omega$ | FS | SI | OI | SO | $SO_\omega$ |
| **CEC 1** — $f(x^*_{OFF})$ = -1.08e4±100.33 | | | | | | | | | | | | | | | |
| CARCOO | - | - | - | - | - | - | - | - | - | - | - | - | - | - | - |
| DEPF | -1.08e4±100.33 | 1.00±0.00 | 1.00±0.00 | 1.00±0.00 | 1.00±0.00 | -1.08e4±100.33 | 1.00±0.00 | 1.00±0.00 | 1.00±0.00 | 1.00±0.00 | -1.08e4±100.33 | 1.00±0.00 | 1.00±0.00 | 1.00±0.00 | 1.00±0.00 |
| DESPF | -1.08e4±100.33 | 1.00±0.00 | 1.00±0.00 | 1.00±0.00 | 1.00±0.00 | -1.08e4±100.33 | 1.00±0.00 | 1.00±0.00 | 1.00±0.00 | 1.00±0.00 | -1.08e4±100.33 | 1.00±0.00 | 1.00±0.00 | 1.00±0.00 | 1.00±0.00 |
| PRIME | - | - | - | - | - | - | - | - | - | - | - | - | - | - | - |
| **CEC 2** — $f(x^*_{OFF})$ = -1.08±0.00 | | | | | | | | | | | | | | | |
| CARCOO | -1.25±0.06 | -15.20±5.62 | -15.20±5.62 | -15.20±5.62 | -15.20±5.62 | -1.25±0.06 | -15.32±5.68 | -15.32±5.68 | -15.32±5.68 | -15.32±5.68 | -1.25±0.06 | -15.37±5.70 | -15.37±5.70 | -15.37±5.70 | -15.37±5.70 |
| DEPF | -1.28±0.01 | -18.61±0.20 | -18.61±0.20 | -18.61±0.20 | -18.61±0.20 | -1.28±0.01 | -18.61±0.37 | -18.61±0.37 | -18.61±0.37 | -18.61±0.37 | -1.28±0.00 | -18.68±0.29 | -18.68±0.29 | -18.68±0.29 | -18.68±0.29 |
| DESPF | -1.28±0.08 | -18.68±0.25 | -18.68±0.25 | -18.68±0.25 | -18.68±0.25 | -1.28±0.08 | -18.79±0.26 | -18.79±0.26 | -18.79±0.26 | -18.79±0.26 | -1.28±0.08 | -18.82±0.26 | -18.82±0.26 | -18.82±0.26 | -18.82±0.26 |
| PRIME | -1.28±0.00 | -18.68±0.25 | -18.68±0.25 | -18.68±0.25 | -18.68±0.25 | -1.28±0.00 | -18.79±0.26 | -18.79±0.26 | -18.79±0.26 | -18.79±0.26 | -1.28±0.00 | -18.82±0.26 | -18.82±0.26 | -18.82±0.26 | -18.82±0.26 |
| **CEC 3** — $f(x^*_{OFF})$ = -2.0±0.00 | | | | | | | | | | | | | | | |
| CARCOO | -2.16±0.28 | -14.11±26.34 | -14.11±26.34 | -14.11±26.34 | -14.11±26.34 | -2.17±0.28 | -14.57±27.29 | -14.57±27.29 | -14.57±27.29 | -14.57±27.29 | -2.26±0.37 | -17.29±27.63 | -17.29±27.63 | -17.29±27.63 | -17.29±27.63 |
| DEPF | -2.98±0.13 | -92.22±7.27 | -92.22±7.27 | -92.22±7.27 | -92.22±7.27 | -2.98±0.13 | -94.41±10.14 | -94.41±10.14 | -94.41±10.14 | -94.41±10.14 | -2.98±0.13 | -95.12±11.10 | -95.12±11.10 | -95.12±11.10 | -95.12±11.10 |
| DESPF | -2.99±0.13 | -94.07±8.95 | -94.07±8.95 | -94.07±8.95 | -94.07±8.95 | -3.00±0.13 | -96.36±10.74 | -96.36±10.74 | -96.36±10.74 | -96.36±10.74 | -3.00±0.13 | -97.18±11.40 | -97.18±11.40 | -97.18±11.40 | -97.18±11.40 |
| PRIME | -2.91±0.02 | -88.55±1.74 | -88.55±1.74 | -88.55±1.74 | -88.55±1.74 | -2.91±0.02 | -89.01±1.75 | -89.01±1.75 | -89.01±1.75 | -89.01±1.75 | -2.91±0.02 | -89.17±1.75 | -89.17±1.75 | -89.17±1.75 | -89.17±1.75 |
| **CEC 4** — $f(x^*_{OFF})$ = -264.43±0.30 | | | | | | | | | | | | | | | |
| CARCOO | -295.40±0.41 | -3.06e3±56.42 | -3.06e3±56.42 | -3.06e3±56.42 | -3.06e3±56.42 | -295.40±0.41 | -3.08e3±56.71 | -3.08e3±56.71 | -3.08e3±56.71 | -3.08e3±56.71 | -295.40±0.41 | -3.09e3±56.81 | -3.09e3±56.81 | -3.09e3±56.81 | -3.09e3±56.81 |
| DEPF | -382.84±0.00 | -1.01e4±139.38 | -1.01e4±139.38 | -1.01e4±139.38 | -1.01e4±139.38 | -382.84±0.00 | -1.10e4±69.90 | -1.10e4±69.90 | -1.10e4±69.90 | -1.10e4±69.90 | -382.84±0.00 | -1.13e4±49.18 | -1.13e4±49.18 | -1.13e4±49.18 | -1.13e4±49.18 |
| DESPF | -382.67±0.31 | -8.83e3±2.76e3 | -8.83e3±2.76e3 | -8.83e3±2.76e3 | -8.83e3±2.76e3 | -382.83±0.02 | -9.71e3±3.04e3 | -9.71e3±3.04e3 | -9.71e3±3.04e3 | -9.71e3±3.04e3 | -382.84±0.01 | -1.00e4±3.14e3 | -1.00e4±3.14e3 | -1.00e4±3.14e3 | -1.00e4±3.14e3 |
| PRIME | -295.40±0.41 | -3.06e3±56.42 | -3.06e3±56.42 | -3.06e3±56.42 | -3.06e3±56.42 | -295.40±0.41 | -3.08e3±56.71 | -3.08e3±56.71 | -3.08e3±56.71 | -3.08e3±56.71 | -295.40±0.41 | -3.09e3±56.81 | -3.09e3±56.81 | -3.09e3±56.81 | -3.09e3±56.81 |
| **CEC 5** — $f(x^*_{OFF})$ = -5.84±0.06 | | | | | | | | | | | | | | | |
| CARCOO | -17.99±0.12 | -2.19±0.27 | -2.26±0.22 | -2.22±0.24 | -2.23±0.24 | -17.99±0.12 | -2.23±0.26 | -2.29±0.21 | -2.26±0.24 | -2.25±0.24 | -17.99±0.12 | -2.24±0.26 | -2.30±0.21 | -2.27±0.23 | -2.24±0.26 |
| DEPF | -18.08±0.31 | -152.59±390.50 | -152.59±390.50 | -152.59±390.50 | -152.59±390.50 | -24.08±14.39 | -153.93±392.33 | -153.93±392.33 | -153.93±392.33 | -153.93±392.33 | -67.28±0.00 | -155.72±392.43 | -155.72±392.43 | -155.72±392.43 | -155.72±392.43 |
| DESPF | -66.32±2.55 | -479.32±1.19e3 | -479.32±1.19e3 | -479.32±1.19e3 | -479.32±1.19e3 | -67.28±0.00 | -648.28±1.61e3 | -648.28±1.61e3 | -648.28±1.61e3 | -648.28±1.61e3 | -67.28±0.00 | -703.90±1.74e3 | -703.90±1.74e3 | -703.90±1.74e3 | -703.90±1.74e3 |
| PRIME | -17.99±0.11 | -2.96±0.30 | -2.96±0.30 | -2.96±0.30 | -2.96±0.30 | -17.99±0.11 | -2.98±0.30 | -2.98±0.30 | -2.98±0.30 | -2.98±0.30 | -17.99±0.11 | -2.98±0.30 | -2.98±0.30 | -2.98±0.30 | -2.98±0.30 |

Table 82: Overall results for CEC constrained tasks with 128 solutions and 50th percentile evaluations. In this case, 30% of the values are missing near the worst value and another 20% near the optimal value. Details are the same as Table 5.

| Steps | t = 50 | | | | | t = 100 | | | | | t = 150 | | | | |
|---|---|---|---|---|---|---|---|---|---|---|---|---|---|---|---|
| Metric | FS | SI | OI | SO | $SO_\omega$ | FS | SI | OI | SO | $SO_\omega$ | FS | SI | OI | SO | $SO_\omega$ |
| **CEC 1** — $f(x^*_{OFF})$ = -9.91e3±100.88 | | | | | | | | | | | | | | | |
| CARCOO | - | - | - | - | - | - | - | - | - | - | - | - | - | - | - |
| DEPF | -9.91e3±100.88 | 1.00±0.00 | 1.00±0.00 | 1.00±0.00 | 1.00±0.00 | -9.91e3±100.88 | 1.00±0.00 | 1.00±0.00 | 1.00±0.00 | 1.00±0.00 | -9.91e3±100.88 | 1.00±0.00 | 1.00±0.00 | 1.00±0.00 | 1.00±0.00 |
| DESPF | -9.91e3±100.88 | 1.00±0.00 | 1.00±0.00 | 1.00±0.00 | 1.00±0.00 | -9.91e3±100.88 | 1.00±0.00 | 1.00±0.00 | 1.00±0.00 | 1.00±0.00 | -9.91e3±100.88 | 1.00±0.00 | 1.00±0.00 | 1.00±0.00 | 1.00±0.00 |
| PRIME | - | - | - | - | - | - | - | - | - | - | - | - | - | - | - |
| **CEC 2** — $f(x^*_{OFF})$ = -1.08±0.00 | | | | | | | | | | | | | | | |
| CARCOO | -1.26±0.00 | -16.45±0.22 | -16.45±0.22 | -16.45±0.22 | -16.45±0.22 | -1.26±0.00 | -16.54±0.22 | -16.54±0.22 | -16.54±0.22 | -16.54±0.22 | -1.26±0.00 | -16.57±0.23 | -16.57±0.23 | -16.57±0.23 | -16.57±0.23 |
| DEPF | -1.26±0.00 | -16.44±0.22 | -16.44±0.22 | -16.44±0.22 | -16.44±0.22 | -1.26±0.00 | -16.49±0.23 | -16.49±0.23 | -16.49±0.23 | -16.49±0.23 | -1.26±0.00 | -16.51±0.24 | -16.51±0.24 | -16.51±0.24 | -16.51±0.24 |
| DESPF | -1.26±0.00 | -16.45±0.23 | -16.45±0.23 | -16.45±0.23 | -16.45±0.23 | -1.26±0.00 | -16.54±0.23 | -16.54±0.23 | -16.54±0.23 | -16.54±0.23 | -1.26±0.00 | -16.57±0.23 | -16.57±0.23 | -16.57±0.23 | -16.57±0.23 |
| PRIME | -1.26±0.00 | -16.45±0.23 | -16.45±0.23 | -16.45±0.23 | -16.45±0.23 | -1.26±0.00 | -16.54±0.23 | -16.54±0.23 | -16.54±0.23 | -16.54±0.23 | -1.26±0.00 | -16.57±0.23 | -16.57±0.23 | -16.57±0.23 | -16.57±0.23 |
| **CEC 3** — $f(x^*_{OFF})$ = -2.0±0.00 | | | | | | | | | | | | | | | |
| CARCOO | -2.18±0.23 | -12.21±11.27 | -12.21±11.27 | -12.21±11.27 | -12.21±11.27 | -2.27±0.31 | -17.00±18.05 | -17.00±18.05 | -17.00±18.05 | -17.00±18.05 | -2.27±0.31 | -19.92±21.86 | -19.92±21.86 | -19.92±21.86 | -19.92±21.86 |
| DEPF | -2.79±0.00 | -77.03±1.97 | -77.03±1.97 | -77.03±1.97 | -77.03±1.97 | -2.79±0.02 | -77.44±1.98 | -77.44±1.98 | -77.44±1.98 | -77.44±1.98 | -2.79±0.02 | -77.57±1.98 | -77.57±1.98 | -77.57±1.98 | -77.57±1.98 |
| DESPF | -2.84±0.14 | -79.31±6.70 | -79.31±6.70 | -79.31±6.70 | -79.31±6.70 | -2.84±0.14 | -81.09±10.25 | -81.09±10.25 | -81.09±10.25 | -81.09±10.25 | -2.84±0.14 | -81.68±11.43 | -81.68±11.43 | -81.68±11.43 | -81.68±11.43 |
| PRIME | -2.79±0.02 | -77.03±1.97 | -77.03±1.97 | -77.03±1.97 | -77.03±1.97 | -2.79±0.02 | -77.44±1.98 | -77.44±1.98 | -77.44±1.98 | -77.44±1.98 | -2.79±0.02 | -77.57±1.98 | -77.57±1.98 | -77.57±1.98 | -77.57±1.98 |
| **CEC 4** — $f(x^*_{OFF})$ = -264.43±0.30 | | | | | | | | | | | | | | | |
| CARCOO | -286.68±0.71 | -2.20e3±71.44 | -2.20e3±71.44 | -2.20e3±71.44 | -2.20e3±71.44 | -286.68±0.71 | -2.21e3±71.82 | -2.21e3±71.82 | -2.21e3±71.82 | -2.21e3±71.82 | -286.68±0.71 | -2.22e3±71.94 | -2.22e3±71.94 | -2.22e3±71.94 | -2.22e3±71.94 |
| DEPF | -379.68±5.65 | -9.53e3±647.09 | -9.53e3±647.09 | -9.53e3±647.09 | -9.53e3±647.09 | -380.74±5.21 | -1.06e4±538.38 | -1.06e4±538.38 | -1.06e4±538.38 | -1.06e4±538.38 | -380.74±5.21 | -1.09e4±519.97 | -1.09e4±519.97 | -1.09e4±519.97 | -1.09e4±519.97 |
| DESPF | -382.75±0.23 | -8.66e3±3.14e3 | -8.66e3±3.14e3 | -8.66e3±3.14e3 | -8.66e3±3.14e3 | -382.84±0.02 | -9.55e3±3.46e3 | -9.55e3±3.46e3 | -9.55e3±3.46e3 | -9.55e3±3.46e3 | -382.84±0.00 | -9.84e3±3.56e3 | -9.84e3±3.56e3 | -9.84e3±3.56e3 | -9.84e3±3.56e3 |
| PRIME | -286.68±0.71 | -2.20e3±71.44 | -2.20e3±71.44 | -2.20e3±71.44 | -2.20e3±71.44 | -286.68±0.71 | -2.21e3±71.82 | -2.21e3±71.82 | -2.21e3±71.82 | -2.21e3±71.82 | -286.68±0.71 | -2.22e3±71.94 | -2.22e3±71.94 | -2.22e3±71.94 | -2.22e3±71.94 |
| **CEC 5** — $f(x^*_{OFF})$ = -4.91±0.00 | | | | | | | | | | | | | | | |
| CARCOO | -16.07±0.24 | -3.25±0.38 | -3.25±0.38 | -3.25±0.38 | -3.25±0.38 | -16.07±0.24 | -3.27±0.39 | -3.27±0.39 | -3.27±0.39 | -3.27±0.39 | -16.07±0.24 | -3.27±0.39 | -3.27±0.39 | -3.27±0.39 | -3.27±0.39 |
| DEPF | -22.12±11.98 | -151.78±374.63 | -151.78±374.63 | -151.78±374.63 | -151.78±374.63 | -44.82±31.36 | -166.28±392.36 | -166.28±392.36 | -166.28±392.36 | -166.28±392.36 | -60.91±12.72 | -204.71±442.38 | -204.71±442.38 | -204.71±442.38 | -204.71±442.38 |
| DESPF | -64.20±4.59 | -528.16±1.25e3 | -528.16±1.25e3 | -528.16±1.25e3 | -528.16±1.25e3 | -66.68±1.59 | -703.44±1.64e3 | -703.44±1.64e3 | -703.44±1.64e3 | -703.44±1.64e3 | -66.68±1.59 | -761.52±1.75e3 | -761.52±1.75e3 | -761.52±1.75e3 | -761.52±1.75e3 |
| PRIME | -16.08±0.23 | -4.53±0.79 | -4.53±0.79 | -4.53±0.79 | -4.53±0.79 | -16.08±0.23 | -4.56±0.79 | -4.56±0.79 | -4.56±0.79 | -4.56±0.79 | -16.08±0.23 | -4.57±0.79 | -4.57±0.79 | -4.57±0.79 | -4.57±0.79 |

Table 83: Overall results for CEC constrained tasks with 128 solutions and 50th percentile evaluations. In this case, 40% of the values are missing near the worst value and another 10% near the optimal value. Details are the same as Table 5.

| Steps | t = 50 | | | | | t = 100 | | | | | t = 150 | | | | |
|---|---|---|---|---|---|---|---|---|---|---|---|---|---|---|---|
| Metric | FS | SI | OI | SO | $SO_\omega$ | FS | SI | OI | SO | $SO_\omega$ | FS | SI | OI | SO | $SO_\omega$ |
| **CEC 1** — $f(x^*_{OFF})$ = -8.100e3±108.86 | | | | | | | | | | | | | | | |
| CARCOO | - | - | - | - | - | - | - | - | - | - | - | - | - | - | - |
| DEPF | -8.10e3±108.86 | 1.00±0.00 | 1.00±0.00 | 1.00±0.00 | 1.00±0.00 | -8.10e3±108.86 | 1.00±0.00 | 1.00±0.00 | 1.00±0.00 | 1.00±0.00 | -8.10e3±108.86 | 1.00±0.00 | 1.00±0.00 | 1.00±0.00 | 1.00±0.00 |
| DESPF | -8.10e3±108.86 | 1.00±0.00 | 1.00±0.00 | 1.00±0.00 | 1.00±0.00 | -8.10e3±108.86 | 1.00±0.00 | 1.00±0.00 | 1.00±0.00 | 1.00±0.00 | -8.10e3±108.86 | 1.00±0.00 | 1.00±0.00 | 1.00±0.00 | 1.00±0.00 |
| PRIME | - | - | - | - | - | - | - | - | - | - | - | - | - | - | - |
| **CEC 2** — $f(x^*_{OFF})$ = -1.08±0.00 | | | | | | | | | | | | | | | |
| CARCOO | -1.22±0.02 | -11.70±0.69 | -11.70±0.69 | -11.70±0.69 | -11.70±0.69 | -1.22±0.05 | -12.21±1.84 | -12.21±1.84 | -12.21±1.84 | -12.21±1.84 | -1.22±0.03 | -12.43±2.35 | -12.43±2.35 | -12.43±2.35 | -12.43±2.35 |
| DEPF | -1.24±0.08 | -12.33±1.29 | -12.33±1.29 | -12.33±1.29 | -12.33±1.29 | -1.23±0.05 | -13.14±2.95 | -13.14±2.95 | -13.14±2.95 | -13.14±2.95 | -1.23±0.05 | -13.38±3.56 | -13.38±3.56 | -13.38±3.56 | -13.38±3.56 |
| DESPF | -1.21±0.00 | -11.50±0.41 | -11.50±0.41 | -11.50±0.41 | -11.50±0.41 | -1.21±0.00 | -11.57±0.41 | -11.57±0.41 | -11.57±0.41 | -11.57±0.41 | -1.21±0.00 | -11.59±0.41 | -11.59±0.41 | -11.59±0.41 | -11.59±0.41 |
| PRIME | -1.21±0.00 | -11.50±0.41 | -11.50±0.41 | -11.50±0.41 | -11.50±0.41 | -1.21±0.00 | -11.57±0.41 | -11.57±0.41 | -11.57±0.41 | -11.57±0.41 | -1.21±0.00 | -11.59±0.41 | -11.59±0.41 | -11.59±0.41 | -11.59±0.41 |
| **CEC 3** — $f(x^*_{OFF})$ = -2.0±0.00 | | | | | | | | | | | | | | | |
| CARCOO | -2.34±0.22 | -29.74±19.74 | -29.74±19.74 | -29.74±19.74 | -29.74±19.74 | -2.39±0.22 | -33.55±19.64 | -33.55±19.64 | -33.55±19.64 | -33.55±19.64 | -2.39±0.22 | -35.11±20.04 | -35.11±20.04 | -35.11±20.04 | -35.11±20.04 |
| DEPF | -2.65±0.21 | -60.81±14.78 | -60.81±14.78 | -60.81±14.78 | -60.81±14.78 | -2.65±0.21 | -62.28±17.87 | -62.28±17.87 | -62.28±17.87 | -62.28±17.87 | -2.65±0.21 | -62.76±18.89 | -62.76±18.89 | -62.76±18.89 | -62.76±18.89 |
| DESPF | -2.66±0.21 | -58.27±8.90 | -58.27±8.90 | -58.27±8.90 | -58.27±8.90 | -2.68±0.20 | -61.98±14.81 | -61.98±14.81 | -61.98±14.81 | -61.98±14.81 | -2.68±0.20 | -63.68±16.62 | -63.68±16.62 | -63.68±16.62 | -63.68±16.62 |
| PRIME | -2.57±0.02 | -54.94±1.94 | -54.94±1.94 | -54.94±1.94 | -54.94±1.94 | -2.57±0.02 | -55.23±1.95 | -55.23±1.95 | -55.23±1.95 | -55.23±1.95 | -2.57±0.02 | -55.33±1.95 | -55.33±1.95 | -55.33±1.95 | -55.33±1.95 |
| **CEC 4** — $f(x^*_{OFF})$ = -264.33±0.20 | | | | | | | | | | | | | | | |
| CARCOO | -300.98±31.43 | -1.28e3±1.76e3 | -1.28e3±1.76e3 | -1.28e3±1.76e3 | -1.28e3±1.76e3 | -319.01±56.57 | -2.06e3±2.76e3 | -2.06e3±2.76e3 | -2.06e3±2.76e3 | -2.06e3±2.76e3 | -318.18±40.13 | -2.47e3±3.09e3 | -2.47e3±3.09e3 | -2.47e3±3.09e3 | -2.47e3±3.09e3 |
| DEPF | -382.73±0.65 | -9.76e3±365.50 | -9.76e3±365.50 | -9.76e3±365.50 | -9.76e3±365.50 | -382.83±0.03 | -1.08e4±180.98 | -1.08e4±180.98 | -1.08e4±180.98 | -1.08e4±180.98 | -382.83±0.03 | -1.12e4±119.30 | -1.12e4±119.30 | -1.12e4±119.30 | -1.12e4±119.30 |
| DESPF | -269.17±0.88 | -477.08±97.49 | -477.08±97.49 | -477.08±97.49 | -477.08±97.49 | -269.17±0.88 | -479.57±98.00 | -479.57±98.00 | -479.57±98.00 | -479.57±98.00 | -269.17±0.88 | -480.39±98.16 | -480.39±98.16 | -480.39±98.16 | -480.39±98.16 |
| PRIME | -269.17±0.88 | -477.08±97.49 | -477.08±97.49 | -477.08±97.49 | -477.08±97.49 | -269.17±0.88 | -479.57±98.00 | -479.57±98.00 | -479.57±98.00 | -479.57±98.00 | -269.17±0.88 | -480.39±98.16 | -480.39±98.16 | -480.39±98.16 | -480.39±98.16 |
| **CEC 5** — $f(x^*_{OFF})$ = -264.33±0.20 | | | | | | | | | | | | | | | |
| CARCOO | -13.06±0.19 | -11.46±2.23 | -11.46±2.23 | -11.46±2.23 | -11.46±2.23 | -13.06±0.19 | -11.52±2.24 | -11.52±2.24 | -11.52±2.24 | -11.52±2.24 | -13.06±0.19 | -11.54±2.25 | -11.54±2.25 | -11.54±2.25 | -11.54±2.25 |
| DEPF | -13.70±1.14 | -806.78±358.34 | -806.78±358.34 | -806.78±358.34 | -806.78±358.34 | -16.83±5.11 | -876.53±416.83 | -876.53±416.83 | -876.53±416.83 | -876.53±416.83 | -38.88±9.57 | -1.31e3±753.29 | -1.31e3±753.29 | -1.31e3±753.29 | -1.31e3±753.29 |
| DESPF | -66.06±2.33 | -2.98e3±1.23e3 | -2.98e3±1.23e3 | -2.98e3±1.23e3 | -2.98e3±1.23e3 | -67.28±0.00 | -4.31e3±1.61e3 | -4.31e3±1.61e3 | -4.31e3±1.61e3 | -4.31e3±1.61e3 | -67.28±0.00 | -4.76e3±1.76e3 | -4.76e3±1.76e3 | -4.76e3±1.76e3 | -4.76e3±1.76e3 |
| PRIME | -13.06±0.19 | -508.84±489.87 | -508.84±489.87 | -508.84±489.87 | -508.84±489.87 | -13.06±0.19 | -511.50±492.42 | -511.50±492.42 | -511.50±492.42 | -511.50±492.42 | -13.06±0.19 | -512.37±493.26 | -512.37±493.26 | -512.37±493.26 | -512.37±493.26 |

Table 84: Overall results for CEC constrained tasks with 128 solutions and 0th percentile evaluations. In this case, 0% of the values are missing near the worst value and another 50% near the optimal value. Details are the same as Table 5.

| Steps | t = 50 | | | | | t = 100 | | | | | t = 150 | | | | |
|---|---|---|---|---|---|---|---|---|---|---|---|---|---|---|---|
| Task $f(x^*_{OFF})$ | CEC 1 -2.08e4±424.97 | | | | | | | | | | | | | | |
| Metric | FS | SI | OI | SO | $SO_\omega$ | FS | SI | OI | SO | $SO_\omega$ | FS | SI | OI | SO | $SO_\omega$ |
| CARCOO | - | - | - | - | - | - | - | - | - | - | - | - | - | - | - |
| DEPF | -2.08e4±424.97 | 1.00±0.00 | 1.00±0.00 | 1.00±0.00 | 1.00±0.00 | -2.08e4±424.97 | 1.00±0.00 | 1.00±0.00 | 1.00±0.00 | 1.00±0.00 | -2.08e4±424.97 | 1.00±0.00 | 1.00±0.00 | 1.00±0.00 | 1.00±0.00 |
| DESPF | -2.08e4±424.97 | 1.00±0.00 | 1.00±0.00 | 1.00±0.00 | 1.00±0.00 | -2.08e4±424.97 | 1.00±0.00 | 1.00±0.00 | 1.00±0.00 | 1.00±0.00 | -2.08e4±424.97 | 1.00±0.00 | 1.00±0.00 | 1.00±0.00 | 1.00±0.00 |
| PRIME | - | - | - | - | - | - | - | - | - | - | - | - | - | - | - |
| Task $f(x^*_{OFF})$ | CEC 2 -1.14±0.01 | | | | | | | | | | | | | | |
| CARCOO | -2.05±0.00 | -71.02±28.90 | -71.02±28.90 | -71.02±28.90 | -71.02±28.90 | -2.05±0.00 | -71.39±29.05 | -71.39±29.05 | -71.39±29.05 | -71.39±29.05 | -2.05±0.00 | -71.52±29.10 | -71.52±29.10 | -71.52±29.10 | -71.52±29.10 |
| DEPF | -2.04±0.00 | -81.90±18.65 | -81.90±18.65 | -81.90±18.65 | -81.90±18.65 | -2.04±0.00 | -81.46±18.57 | -81.46±18.57 | -81.46±18.57 | -81.46±18.57 | -2.04±0.00 | -81.89±18.63 | -81.89±18.63 | -81.89±18.63 | -81.89±18.63 |
| DESPF | -2.04±0.00 | -88.81±0.74 | -88.81±0.74 | -88.81±0.74 | -88.81±0.74 | -2.04±0.00 | -89.28±0.75 | -89.28±0.75 | -89.28±0.75 | -89.28±0.75 | -2.04±0.00 | -89.44±0.75 | -89.44±0.75 | -89.44±0.75 | -89.44±0.75 |
| PRIME | -2.04±0.00 | -16.76±2.21 | -16.76±2.21 | -16.76±2.21 | -16.76±2.21 | -2.04±0.00 | -16.85±2.22 | -16.85±2.22 | -16.85±2.22 | -16.85±2.22 | -2.04±0.00 | -16.88±2.22 | -16.88±2.22 | -16.88±2.22 | -16.88±2.22 |
| Task $f(x^*_{OFF})$ | CEC 3 -2.35±0.02 | | | | | | | | | | | | | | |
| CARCOO | -4.20±0.00 | -15.16±3.44 | -15.16±3.44 | -15.16±3.44 | -15.16±3.44 | -4.20±0.00 | -15.24±3.46 | -15.24±3.46 | -15.24±3.46 | -15.24±3.46 | -4.20±0.00 | -15.27±3.46 | -15.27±3.46 | -15.27±3.46 | -15.27±3.46 |
| DEPF | -4.20±0.00 | -4.66±0.69 | -4.66±0.69 | -4.66±0.69 | -4.66±0.69 | -4.20±0.00 | -4.69±0.70 | -4.69±0.70 | -4.69±0.70 | -4.69±0.70 | -4.20±0.00 | -4.70±0.70 | -4.70±0.70 | -4.70±0.70 | -4.70±0.70 |
| DESPF | -4.20±0.00 | -4.57±0.76 | -4.57±0.76 | -4.57±0.76 | -4.57±0.76 | -4.20±0.00 | -4.60±0.77 | -4.60±0.77 | -4.60±0.77 | -4.60±0.77 | -4.20±0.00 | -4.61±0.77 | -4.61±0.77 | -4.61±0.77 | -4.61±0.77 |
| PRIME | -4.20±0.00 | -4.27±0.36 | -4.27±0.36 | -4.27±0.36 | -4.27±0.36 | -4.20±0.00 | -4.30±0.36 | -4.30±0.36 | -4.30±0.36 | -4.30±0.36 | -4.20±0.00 | -4.31±0.36 | -4.31±0.36 | -4.31±0.36 | -4.31±0.36 |
| Task $f(x^*_{OFF})$ | CEC 4 -264.43±0.30 | | | | | | | | | | | | | | |
| CARCOO | -379.59±1.78 | -1.14e4±195.53 | -1.14e4±195.53 | -1.14e4±195.53 | -1.14e4±195.53 | -379.59±1.78 | -1.15e4±196.54 | -1.15e4±196.54 | -1.15e4±196.54 | -1.15e4±196.54 | -379.59±1.78 | -1.15e4±196.88 | -1.15e4±196.88 | -1.15e4±196.88 | -1.15e4±196.88 |
| DEPF | -382.84±0.00 | -1.17e4±28.81 | -1.17e4±28.81 | -1.17e4±28.81 | -1.17e4±28.81 | -382.84±0.00 | -1.18e4±29.48 | -1.18e4±29.48 | -1.18e4±29.48 | -1.18e4±29.48 | -382.84±0.00 | -1.18e4±29.71 | -1.18e4±29.71 | -1.18e4±29.71 | -1.18e4±29.71 |
| DESPF | -382.84±0.00 | -1.17e4±39.85 | -1.17e4±39.85 | -1.17e4±39.85 | -1.17e4±39.85 | -382.84±0.00 | -1.18e4±33.50 | -1.18e4±33.50 | -1.18e4±33.50 | -1.18e4±33.50 | -382.84±0.00 | -1.18e4±32.00 | -1.18e4±32.00 | -1.18e4±32.00 | -1.18e4±32.00 |
| PRIME | -379.59±1.78 | -1.14e4±195.53 | -1.14e4±195.53 | -1.14e4±195.53 | -1.14e4±195.53 | -379.59±1.78 | -1.15e4±196.54 | -1.15e4±196.54 | -1.15e4±196.54 | -1.15e4±196.54 | -379.59±1.78 | -1.15e4±196.88 | -1.15e4±196.88 | -1.15e4±196.88 | -1.15e4±196.88 |
| Task $f(x^*_{OFF})$ | CEC 5 -10.71±0.18 | | | | | | | | | | | | | | |
| CARCOO | -58.62±3.36 | -4.73±0.37 | -4.73±0.37 | -4.73±0.37 | -4.73±0.37 | -58.62±3.36 | -4.76±0.37 | -4.76±0.37 | -4.76±0.37 | -4.76±0.37 | -58.62±3.36 | -4.77±0.37 | -4.77±0.37 | -4.77±0.37 | -4.77±0.37 |
| DEPF | -63.50±2.99 | -6.18±0.89 | -6.18±0.89 | -6.18±0.89 | -6.18±0.89 | -63.70±2.81 | -6.31±0.90 | -6.31±0.90 | -6.31±0.90 | -6.31±0.90 | -64.86±2.32 | -6.38±0.90 | -6.38±0.90 | -6.38±0.90 | -6.38±0.90 |
| DESPF | -67.28±0.00 | -8.98±1.88 | -8.98±1.88 | -8.98±1.88 | -8.98±1.88 | -67.28±0.00 | -9.09±1.87 | -9.09±1.87 | -9.09±1.87 | -9.09±1.87 | -67.28±0.00 | -9.12±1.87 | -9.12±1.87 | -9.12±1.87 | -9.12±1.87 |
| PRIME | -58.62±3.36 | -5.32±0.49 | -5.32±0.49 | -5.32±0.49 | -5.32±0.49 | -58.62±3.36 | -5.35±0.50 | -5.35±0.50 | -5.35±0.50 | -5.35±0.50 | -58.62±3.36 | -5.36±0.50 | -5.36±0.50 | -5.36±0.50 | -5.36±0.50 |

Table 85: Overall results for CEC constrained tasks with 128 solutions and 0th percentile evaluations. In this case, 0% of the values are missing near the worst value and another 40% near the optimal value. Details are the same as Table 5.

| Steps | t = 50 | | | | | t = 100 | | | | | t = 150 | | | | |
|---|---|---|---|---|---|---|---|---|---|---|---|---|---|---|---|
| Task $f(x^*_{OFF})$ | CEC 1 -2.08e4±424.97 | | | | | | | | | | | | | | |
| Metric | FS | SI | OI | SO | $SO_\omega$ | FS | SI | OI | SO | $SO_\omega$ | FS | SI | OI | SO | $SO_\omega$ |
| CARCOO | - | - | - | - | - | - | - | - | - | - | - | - | - | - | - |
| DEPF | -2.08e4±424.97 | 1.00±0.00 | 1.00±0.00 | 1.00±0.00 | 1.00±0.00 | -2.08e4±424.97 | 1.00±0.00 | 1.00±0.00 | 1.00±0.00 | 1.00±0.00 | -2.08e4±424.97 | 1.00±0.00 | 1.00±0.00 | 1.00±0.00 | 1.00±0.00 |
| DESPF | -2.08e4±424.97 | 1.00±0.00 | 1.00±0.00 | 1.00±0.00 | 1.00±0.00 | -2.08e4±424.97 | 1.00±0.00 | 1.00±0.00 | 1.00±0.00 | 1.00±0.00 | -2.08e4±424.97 | 1.00±0.00 | 1.00±0.00 | 1.00±0.00 | 1.00±0.00 |
| PRIME | - | - | - | - | - | - | - | - | - | - | - | - | - | - | - |
| Task $f(x^*_{OFF})$ | CEC 2 -1.08±0.00 | | | | | | | | | | | | | | |
| CARCOO | -2.04±0.00 | -94.28±0.62 | -94.28±0.62 | -94.28±0.62 | -94.28±0.62 | -2.04±0.00 | -94.78±0.63 | -94.78±0.63 | -94.78±0.63 | -94.78±0.63 | -2.04±0.00 | -94.94±0.63 | -94.94±0.63 | -94.94±0.63 | -94.94±0.63 |
| DEPF | -2.04±0.00 | -94.28±0.62 | -94.28±0.62 | -94.28±0.62 | -94.28±0.62 | -2.04±0.00 | -94.78±0.63 | -94.78±0.63 | -94.78±0.63 | -94.78±0.63 | -2.04±0.00 | -94.94±0.63 | -94.94±0.63 | -94.94±0.63 | -94.94±0.63 |
| DESPF | -2.04±0.00 | -94.28±0.62 | -94.28±0.62 | -94.28±0.62 | -94.28±0.62 | -2.04±0.00 | -94.78±0.63 | -94.78±0.63 | -94.78±0.63 | -94.78±0.63 | -2.04±0.00 | -94.94±0.63 | -94.94±0.63 | -94.94±0.63 | -94.94±0.63 |
| PRIME | -2.04±0.00 | -94.28±0.62 | -94.28±0.62 | -94.28±0.62 | -94.28±0.62 | -2.04±0.00 | -94.78±0.63 | -94.78±0.63 | -94.78±0.63 | -94.78±0.63 | -2.04±0.00 | -94.94±0.63 | -94.94±0.63 | -94.94±0.63 | -94.94±0.63 |
| Task $f(x^*_{OFF})$ | CEC 3 -2.12±0.02 | | | | | | | | | | | | | | |
| CARCOO | -4.20±0.00 | -18.33±2.67 | -18.33±2.67 | -18.33±2.67 | -18.33±2.67 | -4.20±0.00 | -18.43±2.69 | -18.43±2.69 | -18.43±2.69 | -18.43±2.69 | -4.20±0.00 | -18.46±2.69 | -18.46±2.69 | -18.46±2.69 | -18.46±2.69 |
| DEPF | -4.20±0.00 | -17.64±2.82 | -17.64±2.82 | -17.64±2.82 | -17.64±2.82 | -4.20±0.00 | -17.73±2.83 | -17.73±2.83 | -17.73±2.83 | -17.73±2.83 | -4.20±0.00 | -17.77±2.84 | -17.77±2.84 | -17.77±2.84 | -17.77±2.84 |
| DESPF | -4.20±0.00 | -68.73±78.55 | -68.73±78.55 | -68.73±78.55 | -68.73±78.55 | -4.20±0.00 | -69.09±78.94 | -69.09±78.94 | -69.09±78.94 | -69.09±78.94 | -4.20±0.00 | -69.21±79.07 | -69.21±79.07 | -69.21±79.07 | -69.21±79.07 |
| PRIME | -4.20±0.00 | -16.65±2.37 | -16.65±2.37 | -16.65±2.37 | -16.65±2.37 | -4.20±0.00 | -16.74±2.38 | -16.74±2.38 | -16.74±2.38 | -16.74±2.38 | -4.20±0.00 | -16.77±2.39 | -16.77±2.39 | -16.77±2.39 | -16.77±2.39 |
| Task $f(x^*_{OFF})$ | CEC 4 -264.43±0.30 | | | | | | | | | | | | | | |
| CARCOO | -379.59±1.78 | -1.14e4±195.53 | -1.14e4±195.53 | -1.14e4±195.53 | -1.14e4±195.53 | -379.59±1.78 | -1.15e4±196.54 | -1.15e4±196.54 | -1.15e4±196.54 | -1.15e4±196.54 | -379.59±1.78 | -1.15e4±196.88 | -1.15e4±196.88 | -1.15e4±196.88 | -1.15e4±196.88 |
| DEPF | -382.84±0.00 | -8.96e3±4.80e3 | -8.96e3±4.80e3 | -8.96e3±4.80e3 | -8.96e3±4.80e3 | -382.84±0.00 | -9.01e3±4.82e3 | -9.01e3±4.82e3 | -9.01e3±4.82e3 | -9.01e3±4.82e3 | -382.84±0.00 | -9.02e3±4.83e3 | -9.02e3±4.83e3 | -9.02e3±4.83e3 | -9.02e3±4.83e3 |
| DESPF | -382.84±0.01 | -1.03e4±3.61e3 | -1.03e4±3.61e3 | -1.03e4±3.61e3 | -1.03e4±3.61e3 | -382.84±0.00 | -1.04e4±3.65e3 | -1.04e4±3.65e3 | -1.04e4±3.65e3 | -1.04e4±3.65e3 | -382.84±0.00 | -1.04e4±3.64e3 | -1.04e4±3.64e3 | -1.04e4±3.64e3 | -1.04e4±3.64e3 |
| PRIME | -379.59±1.78 | -1.14e4±195.53 | -1.14e4±195.53 | -1.14e4±195.53 | -1.14e4±195.53 | -379.59±1.78 | -1.15e4±196.54 | -1.15e4±196.54 | -1.15e4±196.54 | -1.15e4±196.54 | -379.59±1.78 | -1.15e4±196.88 | -1.15e4±196.88 | -1.15e4±196.88 | -1.15e4±196.88 |
| Task $f(x^*_{OFF})$ | CEC 5 -8.61±0.14 | | | | | | | | | | | | | | |
| CARCOO | -58.62±3.36 | -6.99±0.39 | -6.99±0.39 | -6.99±0.39 | -6.99±0.39 | -58.62±3.36 | -7.03±0.39 | -7.03±0.39 | -7.03±0.39 | -7.03±0.39 | -58.62±3.36 | -7.04±0.39 | -7.04±0.39 | -7.04±0.39 | -7.04±0.39 |
| DEPF | -63.34±3.04 | -10.33±0.76 | -10.33±0.76 | -10.33±0.76 | -10.33±0.76 | -63.83±2.54 | -10.58±0.77 | -10.58±0.77 | -10.58±0.77 | -10.58±0.77 | -65.64±1.81 | -10.74±0.74 | -10.74±0.74 | -10.74±0.74 | -10.74±0.74 |
| DESPF | -67.28±0.00 | -14.02±3.97 | -14.02±3.97 | -14.02±3.97 | -14.02±3.97 | -67.22±0.15 | -14.21±4.02 | -14.21±4.02 | -14.21±4.02 | -14.21±4.02 | -67.28±0.00 | -14.27±4.04 | -14.27±4.04 | -14.27±4.04 | -14.27±4.04 |
| PRIME | -58.62±3.36 | -8.50±0.71 | -8.50±0.71 | -8.50±0.71 | -8.50±0.71 | -58.62±3.36 | -8.55±0.72 | -8.55±0.72 | -8.55±0.72 | -8.55±0.72 | -58.62±3.36 | -8.57±0.72 | -8.57±0.72 | -8.57±0.72 | -8.57±0.72 |

Table 86: Overall results for CEC constrained tasks with 128 solutions and 0th percentile evaluations. In this case, 0% of the values are missing near the worst value and another 30% near the optimal value. Details are the same as Table 5.

| Steps | t = 50 | | | | | t = 100 | | | | | t = 150 | | | | |
|---|---|---|---|---|---|---|---|---|---|---|---|---|---|---|---|
| Task $f(x^*_{OFF})$ | CEC 1 -2.08e4±424.97 | | | | | | | | | | | | | | |
| Metric | FS | SI | OI | SO | $SO_\omega$ | FS | SI | OI | SO | $SO_\omega$ | FS | SI | OI | SO | $SO_\omega$ |
| CARCOO | - | - | - | - | - | - | - | - | - | - | - | - | - | - | - |
| DEPF | -2.08e4±424.97 | 1.00±0.00 | 1.00±0.00 | 1.00±0.00 | 1.00±0.00 | -2.08e4±424.97 | 1.00±0.00 | 1.00±0.00 | 1.00±0.00 | 1.00±0.00 | -2.08e4±424.97 | 1.00±0.00 | 1.00±0.00 | 1.00±0.00 | 1.00±0.00 |
| DESPF | -2.08e4±424.97 | 1.00±0.00 | 1.00±0.00 | 1.00±0.00 | 1.00±0.00 | -2.08e4±424.97 | 1.00±0.00 | 1.00±0.00 | 1.00±0.00 | 1.00±0.00 | -2.08e4±424.97 | 1.00±0.00 | 1.00±0.00 | 1.00±0.00 | 1.00±0.00 |
| PRIME | - | - | - | - | - | - | - | - | - | - | - | - | - | - | - |
| Task $f(x^*_{OFF})$ | CEC 2 -1.08±0.00 | | | | | | | | | | | | | | |
| CARCOO | -2.04±0.00 | -94.05±0.75 | -94.05±0.75 | -94.05±0.75 | -94.05±0.75 | -2.04±0.00 | -94.66±0.63 | -94.66±0.63 | -94.66±0.63 | -94.66±0.63 | -2.04±0.00 | -94.86±0.61 | -94.86±0.61 | -94.86±0.61 | -94.86±0.61 |
| DEPF | -2.04±0.00 | -94.28±0.62 | -94.28±0.62 | -94.28±0.62 | -94.28±0.62 | -1.96±0.23 | -94.73±0.64 | -94.73±0.64 | -94.73±0.64 | -94.73±0.64 | -1.96±0.23 | -91.98±7.69 | -91.98±7.69 | -91.98±7.69 | -91.98±7.69 |
| DESPF | -2.04±0.00 | -94.28±0.62 | -94.28±0.62 | -94.28±0.62 | -94.28±0.62 | -2.04±0.00 | -94.78±0.63 | -94.78±0.63 | -94.78±0.63 | -94.78±0.63 | -2.04±0.00 | -94.94±0.63 | -94.94±0.63 | -94.94±0.63 | -94.94±0.63 |
| PRIME | -2.04±0.00 | -94.28±0.62 | -94.28±0.62 | -94.28±0.62 | -94.28±0.62 | -2.04±0.00 | -94.78±0.63 | -94.78±0.63 | -94.78±0.63 | -94.78±0.63 | -2.04±0.00 | -94.94±0.63 | -94.94±0.63 | -94.94±0.63 | -94.94±0.63 |
| Task $f(x^*_{OFF})$ | CEC 3 -2.0±0.00 | | | | | | | | | | | | | | |
| CARCOO | -4.20±0.00 | -216.35±0.20 | -216.35±0.20 | -216.35±0.20 | -216.35±0.20 | -4.20±0.00 | -217.48±0.20 | -217.48±0.20 | -217.48±0.20 | -217.48±0.20 | -4.20±0.00 | -217.85±0.20 | -217.85±0.20 | -217.85±0.20 | -217.85±0.20 |
| DEPF | -4.20±0.00 | -216.35±0.20 | -216.35±0.20 | -216.35±0.20 | -216.35±0.20 | -4.20±0.00 | -217.48±0.20 | -217.48±0.20 | -217.48±0.20 | -217.48±0.20 | -4.20±0.00 | -217.85±0.20 | -217.85±0.20 | -217.85±0.20 | -217.85±0.20 |
| DESPF | -4.20±0.00 | -216.35±0.20 | -216.35±0.20 | -216.35±0.20 | -216.35±0.20 | -4.20±0.00 | -217.48±0.20 | -217.48±0.20 | -217.48±0.20 | -217.48±0.20 | -4.20±0.00 | -217.85±0.20 | -217.85±0.20 | -217.85±0.20 | -217.85±0.20 |
| PRIME | -4.20±0.00 | -216.35±0.20 | -216.35±0.20 | -216.35±0.20 | -216.35±0.20 | -4.20±0.00 | -217.48±0.20 | -217.48±0.20 | -217.48±0.20 | -217.48±0.20 | -4.20±0.00 | -217.85±0.20 | -217.85±0.20 | -217.85±0.20 | -217.85±0.20 |
| Task $f(x^*_{OFF})$ | CEC 4 -264.43±0.30 | | | | | | | | | | | | | | |
| CARCOO | -379.59±1.78 | -1.14e4±195.53 | -1.14e4±195.53 | -1.14e4±195.53 | -1.14e4±195.53 | -379.59±1.78 | -1.15e4±196.54 | -1.15e4±196.54 | -1.15e4±196.54 | -1.15e4±196.54 | -379.59±1.78 | -1.15e4±196.88 | -1.15e4±196.88 | -1.15e4±196.88 | -1.15e4±196.88 |
| DEPF | -382.84±0.00 | -1.04e4±3.65e3 | -1.04e4±3.65e3 | -1.04e4±3.65e3 | -1.04e4±3.65e3 | -382.84±0.00 | -1.04e4±3.65e3 | -1.04e4±3.65e3 | -1.04e4±3.65e3 | -1.04e4±3.65e3 | -382.84±0.00 | -1.04e4±3.64e3 | -1.04e4±3.64e3 | -1.04e4±3.64e3 | -1.04e4±3.64e3 |
| DESPF | -382.82±0.02 | -7.65e3±3.27e3 | -7.65e3±3.27e3 | -7.65e3±3.27e3 | -7.65e3±3.27e3 | -382.83±0.02 | -7.70e3±3.31e3 | -7.70e3±3.31e3 | -7.70e3±3.31e3 | -7.70e3±3.31e3 | -382.83±0.02 | -7.71e3±3.32e3 | -7.71e3±3.32e3 | -7.71e3±3.32e3 | -7.71e3±3.32e3 |
| PRIME | -379.59±1.78 | -1.01e4±3.55e3 | -1.01e4±3.55e3 | -1.01e4±3.55e3 | -1.01e4±3.55e3 | -379.59±1.78 | -1.01e4±3.54e3 | -1.01e4±3.54e3 | -1.01e4±3.54e3 | -1.01e4±3.54e3 | -379.59±1.78 | -1.01e4±3.55e3 | -1.01e4±3.55e3 | -1.01e4±3.55e3 | -1.01e4±3.55e3 |
| Task $f(x^*_{OFF})$ | CEC 5 -6.75±0.10 | | | | | | | | | | | | | | |
| CARCOO | -58.62±3.36 | -10.82±0.64 | -10.82±0.64 | -10.82±0.64 | -10.82±0.64 | -58.62±3.36 | -10.89±0.65 | -10.89±0.65 | -10.89±0.65 | -10.89±0.65 | -58.62±3.36 | -10.91±0.65 | -10.91±0.65 | -10.91±0.65 | -10.91±0.65 |
| DEPF | -62.65±2.42 | -17.61±6.30 | -17.61±6.30 | -17.61±6.30 | -17.61±6.30 | -62.71±2.38 | -17.88±6.38 | -17.88±6.38 | -17.88±6.38 | -17.88±6.38 | -64.97±2.04 | -18.13±6.45 | -18.13±6.45 | -18.13±6.45 | -18.13±6.45 |
| DESPF | -67.28±0.00 | -109.88±223.59 | -109.88±223.59 | -109.88±223.59 | -109.88±223.59 | -67.28±0.00 | -111.13±225.75 | -111.13±225.75 | -111.13±225.75 | -111.13±225.75 | -67.28±0.00 | -111.54±226.47 | -111.54±226.47 | -111.54±226.47 | -111.54±226.47 |
| PRIME | -58.62±3.36 | -12.40±1.38 | -12.40±1.38 | -12.40±1.38 | -12.40±1.38 | -58.62±3.36 | -12.47±1.39 | -12.47±1.39 | -12.47±1.39 | -12.47±1.39 | -58.62±3.36 | -12.49±1.39 | -12.49±1.39 | -12.49±1.39 | -12.49±1.39 |

Table 87: Overall results for CEC constrained tasks with 128 solutions and 0th percentile evaluations. In this case, 0% of the values are missing near the worst value and another 20% near the optimal value. Details are the same as Table 5.

| Steps | | t = 50 | | | | | t = 100 | | | | | t = 150 | | | |
|---|---|---|---|---|---|---|---|---|---|---|---|---|---|---|---|
| Task $f(x^*_{OFF})$ | | | | | | | CEC 1 $-2.08e4_{\pm424.97}$ | | | | | | | | |
| Metric | FS | SI | OI | SO | $SO_\omega$ | FS | SI | OI | SO | $SO_\omega$ | FS | SI | OI | SO | $SO_\omega$ |
| CARCOO | - | - | - | - | - | - | - | - | - | - | - | - | - | - | - |
| DEPF | $-2.08e4_{\pm424.97}$ | $1.00_{\pm.00}$ | $1.00_{\pm.00}$ | $1.00_{\pm.00}$ | $1.00_{\pm.00}$ | $-2.08e4_{\pm424.97}$ | $1.00_{\pm.00}$ | $1.00_{\pm.00}$ | $1.00_{\pm.00}$ | $1.00_{\pm.00}$ | $-2.08e4_{\pm424.97}$ | $1.00_{\pm.00}$ | $1.00_{\pm.00}$ | $1.00_{\pm.00}$ | $1.00_{\pm.00}$ |
| DESPF | $-2.08e4_{\pm424.97}$ | $1.00_{\pm.00}$ | $1.00_{\pm.00}$ | $1.00_{\pm.00}$ | $1.00_{\pm.00}$ | $-2.08e4_{\pm424.97}$ | $1.00_{\pm.00}$ | $1.00_{\pm.00}$ | $1.00_{\pm.00}$ | $1.00_{\pm.00}$ | $-2.08e4_{\pm424.97}$ | $1.00_{\pm.00}$ | $1.00_{\pm.00}$ | $1.00_{\pm.00}$ | $1.00_{\pm.00}$ |
| PRIME | - | - | - | - | - | - | - | - | - | - | - | - | - | - | - |
| Task $f(x^*_{OFF})$ | | | | | | | CEC 2 $-1.08_{\pm0.00}$ | | | | | | | | |
| CARCOO | $-2.04_{\pm.00}$ | $-94.28_{\pm.62}$ | $-94.28_{\pm.62}$ | $-94.28_{\pm.62}$ | $-94.28_{\pm.62}$ | $-2.04_{\pm.00}$ | $-94.78_{\pm.63}$ | $-94.78_{\pm.63}$ | $-94.78_{\pm.63}$ | $-94.78_{\pm.63}$ | $-2.04_{\pm.00}$ | $-94.94_{\pm.63}$ | $-94.94_{\pm.63}$ | $-94.94_{\pm.63}$ | $-94.94_{\pm.63}$ |
| DEPF | $-2.04_{\pm.00}$ | $-94.28_{\pm.62}$ | $-94.28_{\pm.62}$ | $-94.28_{\pm.62}$ | $-94.28_{\pm.62}$ | $-2.04_{\pm.00}$ | $-94.78_{\pm.63}$ | $-94.78_{\pm.63}$ | $-94.78_{\pm.63}$ | $-94.78_{\pm.63}$ | $-2.04_{\pm.00}$ | $-94.94_{\pm.63}$ | $-94.94_{\pm.63}$ | $-94.94_{\pm.63}$ | $-94.94_{\pm.63}$ |
| DESPF | $-2.04_{\pm.00}$ | $-94.28_{\pm.62}$ | $-94.28_{\pm.62}$ | $-94.28_{\pm.62}$ | $-94.28_{\pm.62}$ | $-2.04_{\pm.00}$ | $-94.78_{\pm.63}$ | $-94.78_{\pm.63}$ | $-94.78_{\pm.63}$ | $-94.78_{\pm.63}$ | $-2.04_{\pm.00}$ | $-94.94_{\pm.63}$ | $-94.94_{\pm.63}$ | $-94.94_{\pm.63}$ | $-94.94_{\pm.63}$ |
| PRIME | $-2.04_{\pm.00}$ | $-94.28_{\pm.62}$ | $-94.28_{\pm.62}$ | $-94.28_{\pm.62}$ | $-94.28_{\pm.62}$ | $-2.04_{\pm.00}$ | $-94.78_{\pm.63}$ | $-94.78_{\pm.63}$ | $-94.78_{\pm.63}$ | $-94.78_{\pm.63}$ | $-2.04_{\pm.00}$ | $-94.94_{\pm.63}$ | $-94.94_{\pm.63}$ | $-94.94_{\pm.63}$ | $-94.94_{\pm.63}$ |
| Task $f(x^*_{OFF})$ | | | | | | | CEC 3 $-2.0_{\pm0.00}$ | | | | | | | | |
| CARCOO | $-4.20_{\pm.00}$ | $-216.35_{\pm.20}$ | $-216.35_{\pm.20}$ | $-216.35_{\pm.20}$ | $-216.35_{\pm.20}$ | $-4.20_{\pm.00}$ | $-217.48_{\pm.20}$ | $-217.48_{\pm.20}$ | $-217.48_{\pm.20}$ | $-217.48_{\pm.20}$ | $-4.20_{\pm.00}$ | $-217.85_{\pm.20}$ | $-217.85_{\pm.20}$ | $-217.85_{\pm.20}$ | $-217.85_{\pm.20}$ |
| DEPF | $-4.20_{\pm.00}$ | $-216.35_{\pm.20}$ | $-216.35_{\pm.20}$ | $-216.35_{\pm.20}$ | $-216.35_{\pm.20}$ | $-4.20_{\pm.00}$ | $-217.48_{\pm.20}$ | $-217.48_{\pm.20}$ | $-217.48_{\pm.20}$ | $-217.48_{\pm.20}$ | $-4.20_{\pm.00}$ | $-217.85_{\pm.20}$ | $-217.85_{\pm.20}$ | $-217.85_{\pm.20}$ | $-217.85_{\pm.20}$ |
| DESPF | $-4.20_{\pm.00}$ | $-216.35_{\pm.20}$ | $-216.35_{\pm.20}$ | $-216.35_{\pm.20}$ | $-216.35_{\pm.20}$ | $-4.20_{\pm.00}$ | $-217.48_{\pm.20}$ | $-217.48_{\pm.20}$ | $-217.48_{\pm.20}$ | $-217.48_{\pm.20}$ | $-4.20_{\pm.00}$ | $-217.85_{\pm.20}$ | $-217.85_{\pm.20}$ | $-217.85_{\pm.20}$ | $-217.85_{\pm.20}$ |
| PRIME | $-4.20_{\pm.00}$ | $-216.35_{\pm.20}$ | $-216.35_{\pm.20}$ | $-216.35_{\pm.20}$ | $-216.35_{\pm.20}$ | $-4.20_{\pm.00}$ | $-217.48_{\pm.20}$ | $-217.48_{\pm.20}$ | $-217.48_{\pm.20}$ | $-217.48_{\pm.20}$ | $-4.20_{\pm.00}$ | $-217.85_{\pm.20}$ | $-217.85_{\pm.20}$ | $-217.85_{\pm.20}$ | $-217.85_{\pm.20}$ |
| Task $f(x^*_{OFF})$ | | | | | | | CEC 4 $-264.43_{\pm0.30}$ | | | | | | | | |
| CARCOO | $-379.59_{\pm1.78}$ | $-1.14e4_{\pm195.53}$ | $-1.14e4_{\pm195.53}$ | $-1.14e4_{\pm195.53}$ | $-1.14e4_{\pm195.53}$ | $-379.59_{\pm1.78}$ | $-1.15e4_{\pm196.54}$ | $-1.15e4_{\pm196.54}$ | $-1.15e4_{\pm196.54}$ | $-1.15e4_{\pm196.54}$ | $-379.59_{\pm1.78}$ | $-1.15e4_{\pm196.88}$ | $-1.15e4_{\pm196.88}$ | $-1.15e4_{\pm196.88}$ | $-1.15e4_{\pm196.88}$ |
| DEPF | $-382.84_{\pm0.00}$ | $-1.04e4_{\pm3.48e3}$ | $-1.04e4_{\pm3.48e3}$ | $-1.04e4_{\pm3.48e3}$ | $-1.04e4_{\pm3.48e3}$ | $-382.84_{\pm0.00}$ | $-1.05e4_{\pm3.49e3}$ | $-1.05e4_{\pm3.49e3}$ | $-1.05e4_{\pm3.49e3}$ | $-1.05e4_{\pm3.49e3}$ | $-382.84_{\pm0.00}$ | $-1.05e4_{\pm3.50e3}$ | $-1.05e4_{\pm3.50e3}$ | $-1.05e4_{\pm3.50e3}$ | $-1.05e4_{\pm3.50e3}$ |
| DESPF | $-382.84_{\pm0.01}$ | $-1.03e4_{\pm3.82e3}$ | $-1.03e4_{\pm3.82e3}$ | $-1.03e4_{\pm3.82e3}$ | $-1.03e4_{\pm3.82e3}$ | $-382.84_{\pm0.00}$ | $-1.03e4_{\pm3.85e3}$ | $-1.03e4_{\pm3.85e3}$ | $-1.03e4_{\pm3.85e3}$ | $-1.03e4_{\pm3.85e3}$ | $-382.84_{\pm0.00}$ | $-1.03e4_{\pm3.85e3}$ | $-1.03e4_{\pm3.85e3}$ | $-1.03e4_{\pm3.85e3}$ | $-1.03e4_{\pm3.85e3}$ |
| PRIME | $-379.59_{\pm1.78}$ | $-9.43e3_{\pm3.73e3}$ | $-9.43e3_{\pm3.73e3}$ | $-9.43e3_{\pm3.73e3}$ | $-9.43e3_{\pm3.73e3}$ | $-379.59_{\pm1.78}$ | $-9.48e3_{\pm3.74e3}$ | $-9.48e3_{\pm3.74e3}$ | $-9.48e3_{\pm3.74e3}$ | $-9.48e3_{\pm3.74e3}$ | $-379.59_{\pm1.78}$ | $-9.49e3_{\pm3.75e3}$ | $-9.49e3_{\pm3.75e3}$ | $-9.49e3_{\pm3.75e3}$ | $-9.49e3_{\pm3.75e3}$ |
| Task $f(x^*_{OFF})$ | | | | | | | CEC 5 $-4.91_{\pm0.30}$ | | | | | | | | |
| CARCOO | $-58.59_{\pm3.59}$ | $-18.70_{\pm2.00}$ | $-18.70_{\pm2.00}$ | $-18.70_{\pm2.00}$ | $-18.70_{\pm2.00}$ | $-58.59_{\pm3.59}$ | $-18.80_{\pm2.01}$ | $-18.80_{\pm2.01}$ | $-18.80_{\pm2.01}$ | $-18.80_{\pm2.01}$ | $-58.59_{\pm3.59}$ | $-18.84_{\pm2.02}$ | $-18.84_{\pm2.02}$ | $-18.84_{\pm2.02}$ | $-18.84_{\pm2.02}$ |
| DEPF | $-64.18_{\pm2.28}$ | $-761.76_{\pm1.88e3}$ | $-761.76_{\pm1.88e3}$ | $-761.76_{\pm1.88e3}$ | $-761.76_{\pm1.88e3}$ | $-64.37_{\pm2.11}$ | $-774.22_{\pm1.91e3}$ | $-774.22_{\pm1.91e3}$ | $-774.22_{\pm1.91e3}$ | $-774.22_{\pm1.91e3}$ | $-66.16_{\pm1.71}$ | $-781.67_{\pm1.93e3}$ | $-781.67_{\pm1.93e3}$ | $-781.67_{\pm1.93e3}$ | $-781.67_{\pm1.93e3}$ |
| DESPF | $-67.28_{\pm0.00}$ | $-2.33e3_{\pm2.94e3}$ | $-2.33e3_{\pm2.94e3}$ | $-2.33e3_{\pm2.94e3}$ | $-2.33e3_{\pm2.94e3}$ | $-67.28_{\pm0.00}$ | $-2.35e3_{\pm2.96e3}$ | $-2.35e3_{\pm2.96e3}$ | $-2.35e3_{\pm2.96e3}$ | $-2.35e3_{\pm2.96e3}$ | $-67.28_{\pm0.00}$ | $-2.36e3_{\pm2.97e3}$ | $-2.36e3_{\pm2.97e3}$ | $-2.36e3_{\pm2.97e3}$ | $-2.36e3_{\pm2.97e3}$ |
| PRIME | $-58.62_{\pm3.36}$ | $-26.59_{\pm4.04}$ | $-26.59_{\pm4.04}$ | $-26.59_{\pm4.04}$ | $-26.59_{\pm4.04}$ | $-58.62_{\pm3.36}$ | $-26.73_{\pm4.06}$ | $-26.73_{\pm4.06}$ | $-26.73_{\pm4.06}$ | $-26.73_{\pm4.06}$ | $-58.62_{\pm3.36}$ | $-26.78_{\pm4.07}$ | $-26.78_{\pm4.07}$ | $-26.78_{\pm4.07}$ | $-26.78_{\pm4.07}$ |

Table 88: Overall results for CEC constrained tasks with 128 solutions and 0th percentile evaluations. In this case, 0% of the values are missing near the worst value and another 10% near the optimal value. Details are the same as Table 5.

| Steps | | t = 50 | | | | | t = 100 | | | | | t = 150 | | | |
|---|---|---|---|---|---|---|---|---|---|---|---|---|---|---|---|
| Task $f(x^*_{OFF})$ | | | | | | | CEC 1 $-2.08e4_{\pm424.97}$ | | | | | | | | |
| Metric | FS | SI | OI | SO | $SO_\omega$ | FS | SI | OI | SO | $SO_\omega$ | FS | SI | OI | SO | $SO_\omega$ |
| CARCOO | - | - | - | - | - | - | - | - | - | - | - | - | - | - | - |
| DEPF | $-2.08e4_{\pm424.97}$ | $1.00_{\pm.00}$ | $1.00_{\pm.00}$ | $1.00_{\pm.00}$ | $1.00_{\pm.00}$ | $-2.08e4_{\pm424.97}$ | $1.00_{\pm.00}$ | $1.00_{\pm.00}$ | $1.00_{\pm.00}$ | $1.00_{\pm.00}$ | $-2.08e4_{\pm424.97}$ | $1.00_{\pm.00}$ | $1.00_{\pm.00}$ | $1.00_{\pm.00}$ | $1.00_{\pm.00}$ |
| DESPF | $-2.08e4_{\pm424.97}$ | $1.00_{\pm.00}$ | $1.00_{\pm.00}$ | $1.00_{\pm.00}$ | $1.00_{\pm.00}$ | $-2.08e4_{\pm424.97}$ | $1.00_{\pm.00}$ | $1.00_{\pm.00}$ | $1.00_{\pm.00}$ | $1.00_{\pm.00}$ | $-2.08e4_{\pm424.97}$ | $1.00_{\pm.00}$ | $1.00_{\pm.00}$ | $1.00_{\pm.00}$ | $1.00_{\pm.00}$ |
| PRIME | - | - | - | - | - | - | - | - | - | - | - | - | - | - | - |
| Task $f(x^*_{OFF})$ | | | | | | | CEC 2 $-1.08_{\pm0.00}$ | | | | | | | | |
| CARCOO | $-2.04_{\pm.00}$ | $-94.28_{\pm.62}$ | $-94.28_{\pm.62}$ | $-94.28_{\pm.62}$ | $-94.28_{\pm.62}$ | $-2.04_{\pm.00}$ | $-94.78_{\pm.63}$ | $-94.78_{\pm.63}$ | $-94.78_{\pm.63}$ | $-94.78_{\pm.63}$ | $-2.04_{\pm.00}$ | $-94.94_{\pm.63}$ | $-94.94_{\pm.63}$ | $-94.94_{\pm.63}$ | $-94.94_{\pm.63}$ |
| DEPF | $-2.04_{\pm.00}$ | $-94.28_{\pm.62}$ | $-94.28_{\pm.62}$ | $-94.28_{\pm.62}$ | $-94.28_{\pm.62}$ | $-2.04_{\pm.00}$ | $-94.78_{\pm.63}$ | $-94.78_{\pm.63}$ | $-94.78_{\pm.63}$ | $-94.78_{\pm.63}$ | $-2.04_{\pm.00}$ | $-94.94_{\pm.63}$ | $-94.94_{\pm.63}$ | $-94.94_{\pm.63}$ | $-94.94_{\pm.63}$ |
| DESPF | $-2.04_{\pm.00}$ | $-94.28_{\pm.62}$ | $-94.28_{\pm.62}$ | $-94.28_{\pm.62}$ | $-94.28_{\pm.62}$ | $-2.04_{\pm.00}$ | $-94.78_{\pm.63}$ | $-94.78_{\pm.63}$ | $-94.78_{\pm.63}$ | $-94.78_{\pm.63}$ | $-2.04_{\pm.00}$ | $-94.94_{\pm.63}$ | $-94.94_{\pm.63}$ | $-94.94_{\pm.63}$ | $-94.94_{\pm.63}$ |
| PRIME | $-2.04_{\pm.00}$ | $-94.28_{\pm.62}$ | $-94.28_{\pm.62}$ | $-94.28_{\pm.62}$ | $-94.28_{\pm.62}$ | $-2.04_{\pm.00}$ | $-94.78_{\pm.63}$ | $-94.78_{\pm.63}$ | $-94.78_{\pm.63}$ | $-94.78_{\pm.63}$ | $-2.04_{\pm.00}$ | $-94.94_{\pm.63}$ | $-94.94_{\pm.63}$ | $-94.94_{\pm.63}$ | $-94.94_{\pm.63}$ |
| Task $f(x^*_{OFF})$ | | | | | | | CEC 3 $-2.0_{\pm0.00}$ | | | | | | | | |
| CARCOO | $-4.20_{\pm.00}$ | $-216.33_{\pm.21}$ | $-216.33_{\pm.21}$ | $-216.33_{\pm.21}$ | $-216.33_{\pm.21}$ | $-4.20_{\pm.00}$ | $-217.47_{\pm.21}$ | $-217.47_{\pm.21}$ | $-217.47_{\pm.21}$ | $-217.47_{\pm.21}$ | $-4.20_{\pm.00}$ | $-217.84_{\pm.21}$ | $-217.84_{\pm.21}$ | $-217.84_{\pm.21}$ | $-217.84_{\pm.21}$ |
| DEPF | $-4.20_{\pm.00}$ | $-216.35_{\pm.20}$ | $-216.35_{\pm.20}$ | $-216.35_{\pm.20}$ | $-216.35_{\pm.20}$ | $-4.20_{\pm.00}$ | $-217.48_{\pm.20}$ | $-217.48_{\pm.20}$ | $-217.48_{\pm.20}$ | $-217.48_{\pm.20}$ | $-4.20_{\pm.00}$ | $-217.85_{\pm.20}$ | $-217.85_{\pm.20}$ | $-217.85_{\pm.20}$ | $-217.85_{\pm.20}$ |
| DESPF | $-4.20_{\pm.00}$ | $-216.35_{\pm.20}$ | $-216.35_{\pm.20}$ | $-216.35_{\pm.20}$ | $-216.35_{\pm.20}$ | $-4.20_{\pm.00}$ | $-217.48_{\pm.20}$ | $-217.48_{\pm.20}$ | $-217.48_{\pm.20}$ | $-217.48_{\pm.20}$ | $-4.20_{\pm.00}$ | $-217.85_{\pm.20}$ | $-217.85_{\pm.20}$ | $-217.85_{\pm.20}$ | $-217.85_{\pm.20}$ |
| PRIME | $-4.20_{\pm.00}$ | $-216.35_{\pm.20}$ | $-216.35_{\pm.20}$ | $-216.35_{\pm.20}$ | $-216.35_{\pm.20}$ | $-4.20_{\pm.00}$ | $-217.48_{\pm.20}$ | $-217.48_{\pm.20}$ | $-217.48_{\pm.20}$ | $-217.48_{\pm.20}$ | $-4.20_{\pm.00}$ | $-217.85_{\pm.20}$ | $-217.85_{\pm.20}$ | $-217.85_{\pm.20}$ | $-217.85_{\pm.20}$ |
| Task $f(x^*_{OFF})$ | | | | | | | CEC 4 $-264.43_{\pm0.30}$ | | | | | | | | |
| CARCOO | $-379.59_{\pm1.78}$ | $-1.14e4_{\pm195.53}$ | $-1.14e4_{\pm195.53}$ | $-1.14e4_{\pm195.53}$ | $-1.14e4_{\pm195.53}$ | $-379.59_{\pm1.78}$ | $-1.15e4_{\pm196.54}$ | $-1.15e4_{\pm196.54}$ | $-1.15e4_{\pm196.54}$ | $-1.15e4_{\pm196.54}$ | $-379.59_{\pm1.78}$ | $-1.15e4_{\pm196.88}$ | $-1.15e4_{\pm196.88}$ | $-1.15e4_{\pm196.88}$ | $-1.15e4_{\pm196.88}$ |
| DEPF | $-382.09_{\pm1.36}$ | $-8.91e3_{\pm4.86e3}$ | $-8.91e3_{\pm4.86e3}$ | $-8.91e3_{\pm4.86e3}$ | $-8.91e3_{\pm4.86e3}$ | $-382.78_{\pm0.16}$ | $-8.96e3_{\pm4.88e3}$ | $-8.96e3_{\pm4.88e3}$ | $-8.96e3_{\pm4.88e3}$ | $-8.96e3_{\pm4.88e3}$ | $-382.84_{\pm0.00}$ | $-8.98e3_{\pm4.89e3}$ | $-8.98e3_{\pm4.89e3}$ | $-8.98e3_{\pm4.89e3}$ | $-8.98e3_{\pm4.89e3}$ |
| DESPF | $-382.80_{\pm0.04}$ | $-1.17e4_{\pm49.10}$ | $-1.17e4_{\pm49.10}$ | $-1.17e4_{\pm49.10}$ | $-1.17e4_{\pm49.10}$ | $-382.80_{\pm0.04}$ | $-1.18e4_{\pm38.70}$ | $-1.18e4_{\pm38.70}$ | $-1.18e4_{\pm38.70}$ | $-1.18e4_{\pm38.70}$ | $-382.80_{\pm0.04}$ | $-1.18e4_{\pm35.61}$ | $-1.18e4_{\pm35.61}$ | $-1.18e4_{\pm35.61}$ | $-1.18e4_{\pm35.61}$ |
| PRIME | $-379.59_{\pm1.78}$ | $-1.01e4_{\pm3.68e3}$ | $-1.01e4_{\pm3.68e3}$ | $-1.01e4_{\pm3.68e3}$ | $-1.01e4_{\pm3.68e3}$ | $-379.59_{\pm1.78}$ | $-1.01e4_{\pm3.70e3}$ | $-1.01e4_{\pm3.70e3}$ | $-1.01e4_{\pm3.70e3}$ | $-1.01e4_{\pm3.70e3}$ | $-379.59_{\pm1.78}$ | $-1.01e4_{\pm3.71e3}$ | $-1.01e4_{\pm3.71e3}$ | $-1.01e4_{\pm3.71e3}$ | $-1.01e4_{\pm3.71e3}$ |
| Task $f(x^*_{OFF})$ | | | | | | | CEC 5 $-2.95_{\pm0.08}$ | | | | | | | | |
| CARCOO | $-58.62_{\pm3.36}$ | $-88.00_{\pm30.42}$ | $-88.00_{\pm30.42}$ | $-88.00_{\pm30.42}$ | $-88.00_{\pm30.42}$ | $-58.62_{\pm3.36}$ | $-88.46_{\pm30.58}$ | $-88.46_{\pm30.58}$ | $-88.46_{\pm30.58}$ | $-88.46_{\pm30.58}$ | $-58.62_{\pm3.36}$ | $-88.62_{\pm30.63}$ | $-88.62_{\pm30.63}$ | $-88.62_{\pm30.63}$ | $-88.62_{\pm30.63}$ |
| DEPF | $-61.82_{\pm2.56}$ | $-5.06e3_{\pm1.65e3}$ | $-5.06e3_{\pm1.65e3}$ | $-5.06e3_{\pm1.65e3}$ | $-5.06e3_{\pm1.65e3}$ | $-61.98_{\pm2.52}$ | $-5.15e3_{\pm1.67e3}$ | $-5.15e3_{\pm1.67e3}$ | $-5.15e3_{\pm1.67e3}$ | $-5.15e3_{\pm1.67e3}$ | $-64.45_{\pm2.63}$ | $-5.21e3_{\pm1.69e3}$ | $-5.21e3_{\pm1.69e3}$ | $-5.21e3_{\pm1.69e3}$ | $-5.21e3_{\pm1.69e3}$ |
| DESPF | $-66.99_{\pm0.78}$ | $-6.20e3_{\pm118.05}$ | $-6.20e3_{\pm118.05}$ | $-6.20e3_{\pm118.05}$ | $-6.20e3_{\pm118.05}$ | $-67.27_{\pm0.04}$ | $-6.32e3_{\pm65.32}$ | $-6.32e3_{\pm65.32}$ | $-6.32e3_{\pm65.32}$ | $-6.32e3_{\pm65.32}$ | $-67.27_{\pm0.04}$ | $-6.35e3_{\pm43.60}$ | $-6.35e3_{\pm43.60}$ | $-6.35e3_{\pm43.60}$ | $-6.35e3_{\pm43.60}$ |
| PRIME | $-58.62_{\pm3.36}$ | $-4.83e3_{\pm1.83e3}$ | $-4.83e3_{\pm1.83e3}$ | $-4.83e3_{\pm1.83e3}$ | $-4.83e3_{\pm1.83e3}$ | $-58.62_{\pm3.36}$ | $-4.85e3_{\pm1.84e3}$ | $-4.85e3_{\pm1.84e3}$ | $-4.85e3_{\pm1.84e3}$ | $-4.85e3_{\pm1.84e3}$ | $-58.62_{\pm3.36}$ | $-4.86e3_{\pm1.85e3}$ | $-4.86e3_{\pm1.85e3}$ | $-4.86e3_{\pm1.85e3}$ | $-4.86e3_{\pm1.85e3}$ |

Table 89: Overall results for CEC constrained tasks with 128 solutions and 0th percentile evaluations. In this case, 10% of the values are missing near the worst value and another 40% near the optimal value. Details are the same as Table 5.

| Steps | | t = 50 | | | | | t = 100 | | | | | t = 150 | | | |
|---|---|---|---|---|---|---|---|---|---|---|---|---|---|---|---|
| Task $f(x^*_{OFF})$ | | | | | | | CEC 1 $-1.39e4_{\pm94.68}$ | | | | | | | | |
| Metric | FS | SI | OI | SO | $SO_\omega$ | FS | SI | OI | SO | $SO_\omega$ | FS | SI | OI | SO | $SO_\omega$ |
| CARCOO | - | - | - | - | - | - | - | - | - | - | - | - | - | - | - |
| DEPF | $-1.39e4_{\pm94.68}$ | $1.00_{\pm.00}$ | $1.00_{\pm.00}$ | $1.00_{\pm.00}$ | $1.00_{\pm.00}$ | $-1.39e4_{\pm94.68}$ | $1.00_{\pm.00}$ | $1.00_{\pm.00}$ | $1.00_{\pm.00}$ | $1.00_{\pm.00}$ | $-1.39e4_{\pm94.68}$ | $1.00_{\pm.00}$ | $1.00_{\pm.00}$ | $1.00_{\pm.00}$ | $1.00_{\pm.00}$ |
| DESPF | $-1.39e4_{\pm94.68}$ | $1.00_{\pm.00}$ | $1.00_{\pm.00}$ | $1.00_{\pm.00}$ | $1.00_{\pm.00}$ | $-1.39e4_{\pm94.68}$ | $1.00_{\pm.00}$ | $1.00_{\pm.00}$ | $1.00_{\pm.00}$ | $1.00_{\pm.00}$ | $-1.39e4_{\pm94.68}$ | $1.00_{\pm.00}$ | $1.00_{\pm.00}$ | $1.00_{\pm.00}$ | $1.00_{\pm.00}$ |
| PRIME | - | - | - | - | - | - | - | - | - | - | - | - | - | - | - |
| Task $f(x^*_{OFF})$ | | | | | | | CEC 2 $-1.08_{\pm0.00}$ | | | | | | | | |
| CARCOO | $-1.34_{\pm.00}$ | $-24.34_{\pm.30}$ | $-24.34_{\pm.30}$ | $-24.34_{\pm.30}$ | $-24.34_{\pm.30}$ | $-1.34_{\pm.00}$ | $-24.47_{\pm.30}$ | $-24.47_{\pm.30}$ | $-24.47_{\pm.30}$ | $-24.47_{\pm.30}$ | $-1.34_{\pm.00}$ | $-24.51_{\pm.30}$ | $-24.51_{\pm.30}$ | $-24.51_{\pm.30}$ | $-24.51_{\pm.30}$ |
| DEPF | $-1.35_{\pm.00}$ | $-25.47_{\pm.33}$ | $-25.47_{\pm.33}$ | $-25.47_{\pm.33}$ | $-25.47_{\pm.33}$ | $-1.34_{\pm.01}$ | $-25.54_{\pm.40}$ | $-25.54_{\pm.40}$ | $-25.54_{\pm.40}$ | $-25.54_{\pm.40}$ | $-1.34_{\pm.01}$ | $-25.45_{\pm.48}$ | $-25.45_{\pm.48}$ | $-25.45_{\pm.48}$ | $-25.45_{\pm.48}$ |
| DESPF | $-1.34_{\pm.00}$ | $-25.17_{\pm.36}$ | $-25.17_{\pm.36}$ | $-25.17_{\pm.36}$ | $-25.17_{\pm.36}$ | $-1.34_{\pm.00}$ | $-25.15_{\pm.51}$ | $-25.15_{\pm.51}$ | $-25.15_{\pm.51}$ | $-25.15_{\pm.51}$ | $-1.34_{\pm.00}$ | $-25.14_{\pm.56}$ | $-25.14_{\pm.56}$ | $-25.14_{\pm.56}$ | $-25.14_{\pm.56}$ |
| PRIME | $-1.34_{\pm.00}$ | $-24.28_{\pm.33}$ | $-24.28_{\pm.33}$ | $-24.28_{\pm.33}$ | $-24.28_{\pm.33}$ | $-1.34_{\pm.00}$ | $-24.41_{\pm.33}$ | $-24.41_{\pm.33}$ | $-24.41_{\pm.33}$ | $-24.41_{\pm.33}$ | $-1.34_{\pm.00}$ | $-24.45_{\pm.33}$ | $-24.45_{\pm.33}$ | $-24.45_{\pm.33}$ | $-24.45_{\pm.33}$ |
| Task $f(x^*_{OFF})$ | | | | | | | CEC 3 $-2.12_{\pm0.02}$ | | | | | | | | |
| CARCOO | $-3.36_{\pm.08}$ | $-24.74_{\pm38.55}$ | $-24.74_{\pm38.55}$ | $-24.74_{\pm38.55}$ | $-24.74_{\pm38.55}$ | $-3.36_{\pm.08}$ | $-24.87_{\pm38.75}$ | $-24.87_{\pm38.75}$ | $-24.87_{\pm38.75}$ | $-24.87_{\pm38.75}$ | $-3.36_{\pm.08}$ | $-24.91_{\pm38.82}$ | $-24.91_{\pm38.82}$ | $-24.91_{\pm38.82}$ | $-24.91_{\pm38.82}$ |
| DEPF | $-3.36_{\pm.08}$ | $-10.75_{\pm1.91}$ | $-10.75_{\pm1.91}$ | $-10.75_{\pm1.91}$ | $-10.75_{\pm1.91}$ | $-3.36_{\pm.08}$ | $-10.81_{\pm1.92}$ | $-10.81_{\pm1.92}$ | $-10.81_{\pm1.92}$ | $-10.81_{\pm1.92}$ | $-3.36_{\pm.08}$ | $-10.83_{\pm1.93}$ | $-10.83_{\pm1.93}$ | $-10.83_{\pm1.93}$ | $-10.83_{\pm1.93}$ |
| DESPF | $-3.36_{\pm.08}$ | $-25.05_{\pm34.45}$ | $-25.05_{\pm34.45}$ | $-25.05_{\pm34.45}$ | $-25.05_{\pm34.45}$ | $-3.36_{\pm.08}$ | $-25.19_{\pm34.63}$ | $-25.19_{\pm34.63}$ | $-25.19_{\pm34.63}$ | $-25.19_{\pm34.63}$ | $-3.36_{\pm.08}$ | $-25.23_{\pm34.69}$ | $-25.23_{\pm34.69}$ | $-25.23_{\pm34.69}$ | $-25.23_{\pm34.69}$ |
| PRIME | $-3.36_{\pm.08}$ | $-9.51_{\pm1.27}$ | $-9.51_{\pm1.27}$ | $-9.51_{\pm1.27}$ | $-9.51_{\pm1.27}$ | $-3.36_{\pm.08}$ | $-9.57_{\pm1.27}$ | $-9.57_{\pm1.27}$ | $-9.57_{\pm1.27}$ | $-9.57_{\pm1.27}$ | $-3.36_{\pm.08}$ | $-9.59_{\pm1.27}$ | $-9.59_{\pm1.27}$ | $-9.59_{\pm1.27}$ | $-9.59_{\pm1.27}$ |
| Task $f(x^*_{OFF})$ | | | | | | | CEC 4 $-264.43_{\pm0.30}$ | | | | | | | | |
| CARCOO | $-327.69_{\pm1.16}$ | $-6.26e3_{\pm110.35}$ | $-6.26e3_{\pm110.35}$ | $-6.26e3_{\pm110.35}$ | $-6.26e3_{\pm110.35}$ | $-327.69_{\pm1.16}$ | $-6.29e3_{\pm110.92}$ | $-6.29e3_{\pm110.92}$ | $-6.29e3_{\pm110.92}$ | $-6.29e3_{\pm110.92}$ | $-327.69_{\pm1.16}$ | $-6.30e3_{\pm111.11}$ | $-6.30e3_{\pm111.11}$ | $-6.30e3_{\pm111.11}$ | $-6.30e3_{\pm111.11}$ |
| DEPF | $-382.84_{\pm0.00}$ | $-1.17e4_{\pm30.45}$ | $-1.17e4_{\pm30.45}$ | $-1.17e4_{\pm30.45}$ | $-1.17e4_{\pm30.45}$ | $-382.84_{\pm0.00}$ | $-1.18e4_{\pm29.31}$ | $-1.18e4_{\pm29.31}$ | $-1.18e4_{\pm29.31}$ | $-1.18e4_{\pm29.31}$ | $-382.84_{\pm0.00}$ | $-1.18e4_{\pm29.39}$ | $-1.18e4_{\pm29.39}$ | $-1.18e4_{\pm29.39}$ | $-1.18e4_{\pm29.39}$ |
| DESPF | $-382.84_{\pm0.01}$ | $-1.05e4_{\pm3.29e3}$ | $-1.05e4_{\pm3.29e3}$ | $-1.05e4_{\pm3.29e3}$ | $-1.05e4_{\pm3.29e3}$ | $-382.84_{\pm0.00}$ | $-1.05e4_{\pm3.30e3}$ | $-1.05e4_{\pm3.30e3}$ | $-1.05e4_{\pm3.30e3}$ | $-1.05e4_{\pm3.30e3}$ | $-382.84_{\pm0.00}$ | $-1.05e4_{\pm3.31e3}$ | $-1.05e4_{\pm3.31e3}$ | $-1.05e4_{\pm3.31e3}$ | $-1.05e4_{\pm3.31e3}$ |
| PRIME | $-327.69_{\pm1.16}$ | $-6.26e3_{\pm110.35}$ | $-6.26e3_{\pm110.35}$ | $-6.26e3_{\pm110.35}$ | $-6.26e3_{\pm110.35}$ | $-327.69_{\pm1.16}$ | $-6.30e3_{\pm124.96}$ | $-6.30e3_{\pm124.96}$ | $-6.30e3_{\pm124.96}$ | $-6.30e3_{\pm124.96}$ | $-327.69_{\pm1.16}$ | $-6.31e3_{\pm120.29}$ | $-6.31e3_{\pm120.29}$ | $-6.31e3_{\pm120.29}$ | $-6.31e3_{\pm120.29}$ |
| Task $f(x^*_{OFF})$ | | | | | | | CEC 5 $-8.61_{\pm0.14}$ | | | | | | | | |
| CARCOO | $-27.28_{\pm0.15}$ | $-2.00_{\pm0.10}$ | $-2.00_{\pm0.10}$ | $-2.00_{\pm0.10}$ | $-2.00_{\pm0.10}$ | $-27.28_{\pm0.15}$ | $-2.02_{\pm0.10}$ | $-2.02_{\pm0.10}$ | $-2.02_{\pm0.10}$ | $-2.02_{\pm0.10}$ | $-27.28_{\pm0.15}$ | $-2.02_{\pm0.10}$ | $-2.02_{\pm0.10}$ | $-2.02_{\pm0.10}$ | $-2.02_{\pm0.10}$ |
| DEPF | $-57.58_{\pm8.25}$ | $-7.94_{\pm1.28}$ | $-7.94_{\pm1.28}$ | $-7.94_{\pm1.28}$ | $-7.94_{\pm1.28}$ | $-59.43_{\pm3.14}$ | $-8.94_{\pm1.47}$ | $-8.94_{\pm1.47}$ | $-8.94_{\pm1.47}$ | $-8.94_{\pm1.47}$ | $-63.40_{\pm4.92}$ | $-9.54_{\pm1.49}$ | $-9.54_{\pm1.49}$ | $-9.54_{\pm1.49}$ | $-9.54_{\pm1.49}$ |
| DESPF | $-67.28_{\pm0.00}$ | $-14.59_{\pm5.09}$ | $-14.59_{\pm5.09}$ | $-14.59_{\pm5.09}$ | $-14.59_{\pm5.09}$ | $-67.28_{\pm0.00}$ | $-14.86_{\pm5.14}$ | $-14.86_{\pm5.14}$ | $-14.86_{\pm5.14}$ | $-14.86_{\pm5.14}$ | $-67.28_{\pm0.00}$ | $-14.95_{\pm5.16}$ | $-14.95_{\pm5.16}$ | $-14.95_{\pm5.16}$ | $-14.95_{\pm5.16}$ |
| PRIME | $-27.27_{\pm0.14}$ | $-2.40_{\pm0.27}$ | $-2.40_{\pm0.27}$ | $-2.40_{\pm0.27}$ | $-2.40_{\pm0.27}$ | $-27.27_{\pm0.14}$ | $-2.42_{\pm0.27}$ | $-2.42_{\pm0.27}$ | $-2.42_{\pm0.27}$ | $-2.42_{\pm0.27}$ | $-27.27_{\pm0.14}$ | $-2.42_{\pm0.27}$ | $-2.42_{\pm0.27}$ | $-2.42_{\pm0.27}$ | $-2.42_{\pm0.27}$ |

Table 90: Overall results for CEC constrained tasks with 128 solutions and 0th percentile evaluations. In this case, 20% of the values are missing near the worst value and another 30% near the optimal value. Details are the same as Table 5.

| Steps | t = 50 | | | | | t = 100 | | | | | t = 150 | | | | |
|---|---|---|---|---|---|---|---|---|---|---|---|---|---|---|---|
| **Task** $f(x^*_{DPF})$ | CEC 1 −1.17e4±74.77 | | | | | | | | | | | | | | |
| Metric CARCOO | FS | SI | OI | SO | SO_ω | FS | SI | OI | SO | SO_ω | FS | SI | OI | SO | SO_ω |
| CARCOO | - | | | | | | | | | | | | | | |
| DEPF | -1.17e4±74.77 | 1.00±0.00 | 1.00±0.00 | 1.00±0.00 | 1.00±0.00 | -1.17e4±74.77 | 1.00±0.00 | 1.00±0.00 | 1.00±0.00 | 1.00±0.00 | -1.17e4±74.77 | 1.00±0.00 | 1.00±0.00 | 1.00±0.00 | 1.00±0.00 |
| DESPF | -1.17e4±74.77 | 1.00±0.00 | 1.00±0.00 | 1.00±0.00 | 1.00±0.00 | -1.17e4±74.77 | 1.00±0.00 | 1.00±0.00 | 1.00±0.00 | 1.00±0.00 | -1.17e4±74.77 | 1.00±0.00 | 1.00±0.00 | 1.00±0.00 | 1.00±0.00 |
| PRIME | - | | | | | | | | | | | | | | |
| **Task** $f(x^*_{DPF})$ | CEC 2 −1.08±0.00 | | | | | | | | | | | | | | |
| CARCOO | -1.31±0.01 | -21.10±0.85 | -21.10±0.85 | -21.10±0.85 | -21.10±0.85 | -1.30±0.00 | -21.06±0.52 | -21.06±0.52 | -21.06±0.52 | -21.06±0.52 | -1.30±0.00 | -21.04±0.38 | -21.04±0.38 | -21.04±0.38 | -21.04±0.38 |
| DEPF | -1.35±0.00 | -25.42±0.36 | -25.42±0.36 | -25.42±0.36 | -25.42±0.36 | -1.35±0.00 | -25.59±0.34 | -25.59±0.34 | -25.59±0.34 | -25.59±0.34 | -1.34±0.02 | -25.45±0.64 | -25.45±0.64 | -25.45±0.64 | -25.45±0.64 |
| DESPF | -1.33±0.02 | -24.83±1.04 | -24.83±1.04 | -24.83±1.04 | -24.83±1.04 | -1.33±0.02 | -24.42±1.69 | -24.42±1.69 | -24.42±1.69 | -24.42±1.69 | -1.33±0.02 | -24.29±1.92 | -24.29±1.92 | -24.29±1.92 | -24.29±1.92 |
| PRIME | -1.30±0.00 | -20.77±0.17 | -20.77±0.17 | -20.77±0.17 | -20.77±0.17 | -1.30±0.00 | -20.86±0.18 | -20.86±0.18 | -20.86±0.18 | -20.86±0.18 | -1.30±0.00 | -20.89±0.19 | -20.89±0.19 | -20.89±0.19 | -20.89±0.19 |
| **Task** $f(x^*_{DPF})$ | CEC 3 −2.0±0.00 | | | | | | | | | | | | | | |
| CARCOO | -3.02±0.01 | -99.77±1.18 | -99.77±1.18 | -99.77±1.18 | -99.77±1.18 | -3.02±0.01 | -100.29±1.18 | -100.29±1.18 | -100.29±1.18 | -100.29±1.18 | -3.02±0.01 | -100.47±1.18 | -100.47±1.18 | -100.47±1.18 | -100.47±1.18 |
| DEPF | -3.20±0.00 | -117.66±0.05 | -117.66±0.05 | -117.66±0.05 | -117.66±0.05 | -3.20±0.00 | -118.28±0.05 | -118.28±0.05 | -118.28±0.05 | -118.28±0.05 | -3.20±0.00 | -118.48±0.06 | -118.48±0.06 | -118.48±0.06 | -118.48±0.06 |
| DESPF | -3.20±0.00 | -117.66±0.05 | -117.66±0.05 | -117.66±0.05 | -117.66±0.05 | -3.20±0.00 | -118.28±0.05 | -118.28±0.05 | -118.28±0.05 | -118.28±0.05 | -3.20±0.00 | -118.48±0.06 | -118.48±0.06 | -118.48±0.06 | -118.48±0.06 |
| PRIME | -3.02±0.00 | -99.83±1.12 | -99.83±1.12 | -99.83±1.12 | -99.83±1.12 | -3.02±0.01 | -100.38±1.10 | -100.38±1.10 | -100.38±1.10 | -100.38±1.10 | -3.02±0.00 | -100.52±1.13 | -100.52±1.13 | -100.52±1.13 | -100.52±1.13 |
| **Task** $f(x^*_{DPF})$ | CEC 4 −264.43±0.30 | | | | | | | | | | | | | | |
| CARCOO | -304.56±0.74 | -3.97e3±82.73 | -3.97e3±82.73 | -3.97e3±82.73 | -3.97e3±82.73 | -304.56±0.74 | -3.99e3±83.16 | -3.99e3±83.16 | -3.99e3±83.16 | -3.99e3±83.16 | -304.56±0.74 | -4.00e3±83.30 | -4.00e3±83.30 | -4.00e3±83.30 | -4.00e3±83.30 |
| DEPF | -382.84±0.00 | -1.17e4±30.45 | -1.17e4±30.45 | -1.17e4±30.45 | -1.17e4±30.45 | -382.84±0.00 | -1.18e4±29.31 | -1.18e4±29.31 | -1.18e4±29.31 | -1.18e4±29.31 | -382.84±0.00 | -1.18e4±29.39 | -1.18e4±29.39 | -1.18e4±29.39 | -1.18e4±29.39 |
| DESPF | -382.83±0.03 | -1.13e4±1.11e3 | -1.13e4±1.11e3 | -1.13e4±1.11e3 | -1.13e4±1.11e3 | -382.83±0.02 | -1.13e4±1.13e3 | -1.13e4±1.13e3 | -1.13e4±1.13e3 | -1.13e4±1.13e3 | -382.83±0.02 | -1.14e4±1.13e3 | -1.14e4±1.13e3 | -1.14e4±1.13e3 | -1.14e4±1.13e3 |
| PRIME | -304.56±0.74 | -3.97e3±82.73 | -3.97e3±82.73 | -3.97e3±82.73 | -3.97e3±82.73 | -304.56±0.74 | -4.01e3±101.79 | -4.01e3±101.79 | -4.01e3±101.79 | -4.01e3±101.79 | -304.56±0.74 | -4.01e3±100.13 | -4.01e3±100.13 | -4.01e3±100.13 | -4.01e3±100.13 |
| **Task** $f(x^*_{DPF})$ | CEC 5 −6.75±0.10 | | | | | | | | | | | | | | |
| CARCOO | - | | | | | | | | | | | | | | |
| DEPF | -48.47±6.33 | -12.24±3.44 | -12.24±3.44 | -12.24±3.44 | -12.24±3.44 | -57.67±8.90 | -14.11±4.19 | -14.11±4.19 | -14.11±4.19 | -14.11±4.19 | -62.52±5.00 | -15.69±4.74 | -15.69±4.74 | -15.69±4.74 | -15.69±4.74 |
| DESPF | -67.28±0.00 | -54.69±69.71 | -54.69±69.71 | -54.69±69.71 | -54.69±69.71 | -67.28±0.00 | -55.59±70.58 | -55.59±70.58 | -55.59±70.58 | -55.59±70.58 | -67.28±0.00 | -55.89±70.86 | -55.89±70.86 | -55.89±70.86 | -55.89±70.86 |
| PRIME | -20.31±0.22 | -2.51±0.30 | -2.51±0.30 | -2.51±0.30 | -2.51±0.30 | -20.31±0.22 | -2.53±0.30 | -2.53±0.30 | -2.53±0.30 | -2.53±0.30 | -20.31±0.22 | -2.53±0.30 | -2.53±0.30 | -2.53±0.30 | -2.53±0.30 |

Table 91: Overall results for CEC constrained tasks with 128 solutions and 0th percentile evaluations. In this case, 25% of the values are missing near the worst value and another 25% near the optimal value. Details are the same as Table 5.

| Steps | t − 50 | | | | | t − 100 | | | | | t − 150 | | | | |
|---|---|---|---|---|---|---|---|---|---|---|---|---|---|---|---|
| **Task** $f(x^*_{DPF})$ | CEC 1 −1.08e4±100.33 | | | | | | | | | | | | | | |
| Metric | FS | SI | OI | SO | SO_ω | FS | SI | OI | SO | SO_ω | FS | SI | OI | SO | SO_ω |
| CARCOO | - | | | | | | | | | | | | | | |
| DEPF | -1.08e4.00±100.33 | 1.00±0.00 | 1.00±0.00 | 1.00±0.00 | 1.00±0.00 | -1.08e4.00±100.33 | 1.00±0.00 | 1.00±0.00 | 1.00±0.00 | 1.00±0.00 | -1.08e4.00±100.33 | 1.00±0.00 | 1.00±0.00 | 1.00±0.00 | 1.00±0.00 |
| DESPF | -1.08e4.00±100.33 | 1.00±0.00 | 1.00±0.00 | 1.00±0.00 | 1.00±0.00 | -1.08e4.00±100.33 | 1.00±0.00 | 1.00±0.00 | 1.00±0.00 | 1.00±0.00 | -1.08e4.00±100.33 | 1.00±0.00 | 1.00±0.00 | 1.00±0.00 | 1.00±0.00 |
| PRIME | - | | | | | | | | | | | | | | |
| **Task** $f(x^*_{DPF})$ | CEC 2 −1.08±0.00 | | | | | | | | | | | | | | |
| CARCOO | -1.28±0.05 | -18.30±0.46 | -18.30±0.46 | -18.30±0.46 | -18.30±0.46 | -1.28±0.00 | -18.50±0.63 | -18.50±0.63 | -18.50±0.63 | -18.50±0.63 | -1.28±0.00 | -18.57±0.49 | -18.57±0.49 | -18.57±0.49 | -18.57±0.49 |
| DEPF | -1.35±0.00 | -25.30±0.45 | -25.30±0.45 | -25.30±0.45 | -25.30±0.45 | -1.34±0.02 | -25.16±1.24 | -25.16±1.24 | -25.16±1.24 | -25.16±1.24 | -1.32±0.00 | -24.96±1.40 | -24.96±1.40 | -24.96±1.40 | -24.96±1.40 |
| DESPF | -1.31±0.03 | -23.37±2.29 | -23.37±2.29 | -23.37±2.29 | -23.37±2.29 | -1.29±0.02 | -21.90±2.10 | -21.90±2.10 | -21.90±2.10 | -21.90±2.10 | -1.30±0.00 | -21.36±1.06 | -21.36±1.06 | -21.36±1.06 | -21.36±1.06 |
| PRIME | -1.28±0.00 | -18.76±0.26 | -18.76±0.26 | -18.76±0.26 | -18.76±0.26 | -1.28±0.00 | -18.82±0.25 | -18.82±0.25 | -18.82±0.25 | -18.82±0.25 | -1.28±0.00 | -18.84±0.25 | -18.84±0.25 | -18.84±0.25 | -18.84±0.25 |
| **Task** $f(x^*_{DPF})$ | CEC 3 −2.00±0.00 | | | | | | | | | | | | | | |
| CARCOO | -2.91±0.02 | -87.05±3.37 | -87.05±3.37 | -87.05±3.37 | -87.05±3.37 | -2.91±0.00 | -88.27±3.23 | -88.27±3.23 | -88.27±3.23 | -88.27±3.23 | -2.91±0.02 | -88.67±1.96 | -88.67±1.96 | -88.67±1.96 | -88.67±1.96 |
| DEPF | -3.20±0.01 | -117.49±0.29 | -117.49±0.29 | -117.49±0.29 | -117.49±0.29 | -3.18±0.00 | -117.71±0.95 | -117.71±0.95 | -117.71±0.95 | -117.71±0.95 | -3.17±0.00 | -117.27±1.45 | -117.27±1.45 | -117.27±1.45 | -117.27±1.45 |
| DESPF | -3.20±0.00 | -117.65±0.08 | -117.65±0.08 | -117.65±0.08 | -117.65±0.08 | -3.19±0.01 | -117.94±0.57 | -117.94±0.57 | -117.94±0.57 | -117.94±0.57 | -3.18±0.02 | -117.82±0.88 | -117.82±0.88 | -117.82±0.88 | -117.82±0.88 |
| PRIME | -2.92±0.01 | -88.56±1.73 | -88.56±1.73 | -88.56±1.73 | -88.56±1.73 | -2.91±0.00 | -89.30±1.84 | -89.30±1.84 | -89.30±1.84 | -89.30±1.84 | -2.91±0.02 | -89.36±1.81 | -89.36±1.81 | -89.36±1.81 | -89.36±1.81 |
| **Task** $f(x^*_{DPF})$ | CEC 4 −264.43±0.06 | | | | | | | | | | | | | | |
| CARCOO | -295.40±0.61 | -3.06e3.00±56.42 | -3.06e3.00±56.42 | -3.06e3.00±56.42 | -3.06e3.00±56.42 | -295.40±0.61 | -3.08e3.00±56.71 | -3.08e3.00±56.71 | -3.08e3.00±56.71 | -3.08e3.00±56.71 | -295.40±0.61 | -3.09e3.00±56.81 | -3.09e3.00±56.81 | -3.09e3.00±56.81 | -3.09e3.00±56.81 |
| DEPF | -382.84±0.00 | -1.17e4.00±30.45 | -1.17e4.00±30.45 | -1.17e4.00±30.45 | -1.17e4.00±30.45 | -382.84±0.00 | -1.18e4.00±29.31 | -1.18e4.00±29.31 | -1.18e4.00±29.31 | -1.18e4.00±29.31 | -382.84±0.00 | -1.18e4.00±29.39 | -1.18e4.00±29.39 | -1.18e4.00±29.39 | -1.18e4.00±29.39 |
| DESPF | -382.84±0.01 | -1.04e4.00±1.36e3 | -1.04e4.00±1.36e3 | -1.04e4.00±1.36e3 | -1.04e4.00±1.36e3 | -382.84±0.01 | -1.05e4.00±1.36e3 | -1.05e4.00±1.36e3 | -1.05e4.00±1.36e3 | -1.05e4.00±1.36e3 | -382.84±0.00 | -1.05e4.00±1.35e3 | -1.05e4.00±1.35e3 | -1.05e4.00±1.35e3 | -1.05e4.00±1.35e3 |
| PRIME | -295.40±0.61 | -3.06e3.00±56.42 | -3.06e3.00±56.42 | -3.06e3.00±56.42 | -3.06e3.00±56.42 | -3.12e3.00±109.71 | -3.12e3.00±109.71 | -3.12e3.00±109.71 | -3.12e3.00±109.71 | -3.12e3.00±109.71 | -295.40±0.61 | -3.11e3.00±95.91 | -3.11e3.00±95.91 | -3.11e3.00±95.91 | -3.11e3.00±95.91 |
| **Task** $f(x^*_{DPF})$ | CEC 5 −5.84±0.06 | | | | | | | | | | | | | | |
| CARCOO | -17.99±0.12 | -2.29±0.21 | -2.29±0.21 | -2.29±0.21 | -2.29±0.21 | -17.99±0.12 | -2.31±0.21 | -2.31±0.21 | -2.31±0.21 | -2.31±0.21 | -17.99±0.12 | -2.31±0.21 | -2.31±0.21 | -2.31±0.21 | -2.31±0.21 |
| DEPF | -38.83±0.31 | -260.18±630.45 | -260.18±630.45 | -260.18±630.45 | -260.18±630.45 | -49.11±7.18 | -263.09±652.90 | -263.09±652.90 | -263.09±652.90 | -263.09±652.90 | -54.37±3.54 | -347.73±670.70 | -347.73±670.70 | -347.73±670.70 | -347.73±670.70 |
| DESPF | -67.28±0.00 | -789.66±1.95e3.00 | -789.66±1.95e3.00 | -789.66±1.95e3.00 | -789.66±1.95e3.00 | -67.28±0.00 | -801.97±1.96e3.00 | -801.97±1.96e3.00 | -801.97±1.96e3.00 | -801.97±1.96e3.00 | -67.28±0.00 | -806.01±1.96e3.00 | -806.01±1.96e3.00 | -806.01±1.96e3.00 | -806.01±1.96e3.00 |
| PRIME | -17.99±0.11 | -2.96±0.30 | -2.96±0.30 | -2.96±0.30 | -2.96±0.30 | -17.99±0.11 | -2.98±0.36 | -2.98±0.36 | -2.98±0.36 | -2.98±0.36 | -17.99±0.11 | -2.98±0.30 | -2.98±0.30 | -2.98±0.30 | -2.98±0.30 |

Table 92: Overall results for CEC constrained tasks with 128 solutions and 0th percentile evaluations. In this case, 30% of the values are missing near the worst value and another 20% near the optimal value. Details are the same as Table 5.

| Steps | t − 50 | | | | | t − 100 | | | | | t − 150 | | | | |
|---|---|---|---|---|---|---|---|---|---|---|---|---|---|---|---|
| **Task** $f(x^*_{DPF})$ | CEC 1 −9.91e3±100.88 | | | | | | | | | | | | | | |
| Metric | FS | SI | OI | SO | SO_ω | FS | SI | OI | SO | SO_ω | FS | SI | OI | SO | SO_ω |
| CARCOO | - | | | | | | | | | | | | | | |
| DEPF | -9.91e3.00±100.88 | 1.00±0.00 | 1.00±0.00 | 1.00±0.00 | 1.00±0.00 | -9.91e3.00±100.88 | 1.00±0.00 | 1.00±0.00 | 1.00±0.00 | 1.00±0.00 | -9.91e3.00±100.88 | 1.00±0.00 | 1.00±0.00 | 1.00±0.00 | 1.00±0.00 |
| DESPF | -9.91e3.00±100.88 | 1.00±0.00 | 1.00±0.00 | 1.00±0.00 | 1.00±0.00 | -9.91e3.00±100.88 | 1.00±0.00 | 1.00±0.00 | 1.00±0.00 | 1.00±0.00 | -9.91e3.00±100.88 | 1.00±0.00 | 1.00±0.00 | 1.00±0.00 | 1.00±0.00 |
| PRIME | - | | | | | | | | | | | | | | |
| **Task** $f(x^*_{DPF})$ | CEC 2 −1.08±0.00 | | | | | | | | | | | | | | |
| CARCOO | -1.26±0.00 | -16.57±0.16 | -16.57±0.16 | -16.57±0.16 | -16.57±0.16 | -1.26±0.00 | -16.60±0.17 | -16.60±0.17 | -16.60±0.17 | -16.60±0.17 | -1.26±0.00 | -16.61±0.09 | -16.61±0.09 | -16.61±0.09 | -16.61±0.09 |
| DEPF | -1.35±0.00 | -25.31±0.39 | -25.31±0.39 | -25.31±0.39 | -25.31±0.39 | -1.34±0.03 | -25.23±1.06 | -25.23±1.06 | -25.23±1.06 | -25.23±1.06 | -1.34±0.00 | -25.04±1.75 | -25.04±1.75 | -25.04±1.75 | -25.04±1.75 |
| DESPF | -1.33±0.04 | -23.69±2.47 | -23.69±2.47 | -23.69±2.47 | -23.69±2.47 | -1.30±0.08 | -22.77±3.27 | -22.77±3.27 | -22.77±3.27 | -22.77±3.27 | -1.30±0.00 | -22.01±3.46 | -22.01±3.46 | -22.01±3.46 | -22.01±3.46 |
| PRIME | -1.26±0.00 | -17.23±0.54 | -17.23±0.54 | -17.23±0.54 | -17.23±0.54 | -1.26±0.00 | -16.92±0.35 | -16.92±0.35 | -16.92±0.35 | -16.92±0.35 | -1.26±0.00 | -16.82±0.29 | -16.82±0.29 | -16.82±0.29 | -16.82±0.29 |
| **Task** $f(x^*_{DPF})$ | CEC 3 −2.00±0.00 | | | | | | | | | | | | | | |
| CARCOO | -2.71±0.21 | -76.49±2.60 | -76.49±2.60 | -76.49±2.60 | -76.49±2.60 | -2.71±0.00 | -73.27±11.53 | -73.27±11.53 | -73.27±11.53 | -73.27±11.53 | -2.71±0.02 | -72.18±14.71 | -72.18±14.71 | -72.18±14.71 | -72.18±14.71 |
| DEPF | -3.08±0.14 | -110.12±7.60 | -110.12±7.60 | -110.12±7.60 | -110.12±7.60 | -3.07±0.16 | -108.37±9.91 | -108.37±9.91 | -108.37±9.91 | -108.37±9.91 | -3.07±0.16 | -107.63±12.05 | -107.63±12.05 | -107.63±12.05 | -107.63±12.05 |
| DESPF | -3.11±0.14 | -110.58±0.48 | -110.58±0.48 | -110.58±0.48 | -110.58±0.48 | -3.04±0.17 | -108.00±11.76 | -108.00±11.76 | -108.00±11.76 | -108.00±11.76 | -3.02±0.16 | -106.01±13.09 | -106.01±13.09 | -106.01±13.09 | -106.01±13.09 |
| PRIME | -2.84±0.12 | -77.31±1.71 | -77.31±1.71 | -77.31±1.71 | -77.31±1.71 | -2.79±0.02 | -77.98±1.72 | -77.98±1.72 | -77.98±1.72 | -77.98±1.72 | -2.79±0.02 | -77.93±1.79 | -77.93±1.79 | -77.93±1.79 | -77.93±1.79 |
| **Task** $f(x^*_{DPF})$ | CEC 4 −264.43±0.04 | | | | | | | | | | | | | | |
| CARCOO | -286.68±0.71 | -2.20e3.00±71.44 | -2.20e3.00±71.44 | -2.20e3.00±71.44 | -2.20e3.00±71.44 | -286.68±0.71 | -2.21e3.00±71.42 | -2.21e3.00±71.42 | -2.21e3.00±71.42 | -2.21e3.00±71.42 | -286.68±0.71 | -2.22e3.00±71.90 | -2.22e3.00±71.90 | -2.22e3.00±71.90 | -2.22e3.00±71.90 |
| DEPF | -381.15±0.40 | -1.15e4.00±313.73 | -1.15e4.00±313.73 | -1.15e4.00±313.73 | -1.15e4.00±313.73 | -381.63±1.15 | -1.16e4.00±283.63 | -1.16e4.00±283.63 | -1.16e4.00±283.63 | -1.16e4.00±283.63 | -380.87±1.15 | -1.16e4.00±160.62 | -1.16e4.00±160.62 | -1.16e4.00±160.62 | -1.16e4.00±160.62 |
| DESPF | -382.84±0.00 | -1.03e4.00±3.75e3.00 | -1.03e4.00±3.75e3.00 | -1.03e4.00±3.75e3.00 | -1.03e4.00±3.75e3.00 | -286.68±0.71 | -1.04e4.00±3.75e3.00 | -1.04e4.00±3.75e3.00 | -1.04e4.00±3.75e3.00 | -1.04e4.00±3.75e3.00 | -382.84±0.00 | -1.04e4.00±3.76e3.00 | -1.04e4.00±3.76e3.00 | -1.04e4.00±3.76e3.00 | -1.04e4.00±3.76e3.00 |
| PRIME | -286.68±0.73 | -2.20e3.00±71.44 | -2.20e3.00±71.44 | -2.20e3.00±71.44 | -2.20e3.00±71.44 | -286.68±0.71 | -2.24e3.00±62.49 | -2.24e3.00±62.49 | -2.24e3.00±62.49 | -2.24e3.00±62.49 | -286.68±0.71 | -2.23e3.00±77.69 | -2.23e3.00±77.89 | -2.23e3.00±77.89 | -2.23e3.00±77.89 |
| **Task** $f(x^*_{DPF})$ | CEC 5 −4.91±0.03 | | | | | | | | | | | | | | |
| CARCOO | -16.07±0.24 | -3.25±0.38 | -3.25±0.38 | -3.25±0.38 | -3.25±0.38 | -16.07±0.00 | -3.27±0.38 | -3.27±0.38 | -3.27±0.38 | -3.27±0.38 | -16.07±0.00 | -3.27±0.38 | -3.27±0.38 | -3.27±0.38 | -3.27±0.38 |
| DEPF | -42.01±0.00 | -289.77±676.74 | -289.77±676.74 | -289.77±676.74 | -289.77±676.74 | -56.43±3.22 | -290.44±656.49 | -290.44±656.49 | -290.44±656.49 | -290.44±656.49 | -63.11±9.06 | -326.94±741.60 | -326.94±741.60 | -326.94±741.60 | -326.94±741.60 |
| DESPF | -67.28±0.03 | -855.76±1.98e3.00 | -855.76±1.98e3.00 | -855.76±1.98e3.00 | -855.76±1.98e3.00 | -66.71±1.51 | -866.75±2.01e3.00 | -866.75±2.01e3.00 | -866.75±2.01e3.00 | -866.75±2.01e3.00 | -66.74±1.44 | -870.10±2.01e3.00 | -870.10±2.01e3.00 | -870.10±2.01e3.00 | -870.10±2.01e3.00 |
| PRIME | -16.08±0.23 | -4.53±0.79 | -4.53±0.79 | -4.53±0.79 | -4.53±0.79 | -16.08±0.23 | -4.56±0.79 | -4.56±0.79 | -4.56±0.79 | -4.56±0.79 | -16.08±0.23 | -4.57±0.79 | -4.57±0.79 | -4.57±0.79 | -4.57±0.79 |

Table 93: Overall results for CEC constrained tasks with 128 solutions and 0th percentile evaluations. In this case, 40% of the values are missing near the worst value and another 10% near the optimal value. Details are the same as Table 5.

| Steps | t − 50 | | | | | t − 100 | | | | | t − 150 | | | | |
|---|---|---|---|---|---|---|---|---|---|---|---|---|---|---|---|
| Task | | | | | | | | CEC 1 | | | | | | | |
| $f(x^*_{DEPF})$ | | | | | | | | −8.10e3±108.86 | | | | | | | |
| Metric | FS | SI | OI | SO | SO$_{ac}$ | FS | SI | OI | SO | SO$_{ac}$ | FS | SI | OI | SO | SO$_{ac}$ |
| CARCOO | | | | | | | | | | | | | | | |
| DEPF | −8.10e3.00±108.86 | 1.00±0.00 | 1.00±0.00 | 1.00±0.00 | 1.00±0.00 | −8.10e3.00±108.86 | 1.00±0.00 | 1.00±0.00 | 1.00±0.00 | 1.00±0.00 | −8.10e3.00±108.86 | 1.00±0.00 | 1.00±0.00 | 1.00±0.00 | 1.00±0.00 |
| DESPF | −8.10e3.00±108.86 | 1.00±0.00 | 1.00±0.00 | 1.00±0.00 | 1.00±0.00 | −8.10e3.00±108.86 | 1.00±0.00 | 1.00±0.00 | 1.00±0.00 | 1.00±0.00 | −8.10e3.00±108.86 | 1.00±0.00 | 1.00±0.00 | 1.00±0.00 | 1.00±0.00 |
| PRIME | | | | | | | | | | | | | | | |
| Task | | | | | | | | CEC 2 | | | | | | | |
| $f(x^*_{DEPF})$ | | | | | | | | −1.08±0.00 | | | | | | | |
| CARCOO | −1.22±0.03 | −12.51±1.96 | −12.51±1.96 | −12.51±1.96 | −12.51±1.96 | −1.22±0.03 | −12.66±2.56 | −12.66±2.56 | −12.66±2.56 | −12.66±2.56 | −1.22±0.03 | −12.73±2.84 | −12.73±2.84 | −12.73±2.84 | −12.73±2.84 |
| DEPF | −1.35±0.00 | −25.15±0.68 | −25.15±0.68 | −25.15±0.68 | −25.15±0.68 | −1.30±0.06 | −24.17±2.34 | −24.17±2.34 | −24.17±2.34 | −24.17±2.34 | −1.28±0.07 | −22.61±3.86 | −22.61±3.86 | −22.61±3.86 | −22.61±3.86 |
| DESPF | −1.28±0.07 | −21.64±3.56 | −21.64±3.56 | −21.64±3.56 | −21.64±3.56 | −1.25±0.06 | −19.48±4.48 | −19.48±4.48 | −19.48±4.48 | −19.48±4.48 | −1.25±0.06 | −18.39±4.75 | −18.39±4.75 | −18.39±4.75 | −18.39±4.75 |
| PRIME | −1.21±0.05 | −12.17±0.08 | −12.17±0.08 | −12.17±0.08 | −12.17±0.08 | −1.21±0.00 | −11.90±0.37 | −11.90±0.37 | −11.90±0.37 | −11.90±0.37 | −1.21±0.00 | −11.81±0.37 | −11.81±0.37 | −11.81±0.37 | −11.81±0.37 |
| Task | | | | | | | | CEC 3 | | | | | | | |
| $f(x^*_{DEPF})$ | | | | | | | | −2.00±0.00 | | | | | | | |
| CARCOO | −2.56±0.02 | −54.82±2.04 | −54.82±2.04 | −54.82±2.04 | −54.82±2.04 | −2.56±0.02 | −55.11±2.86 | −55.11±2.86 | −55.11±2.86 | −55.11±2.86 | −2.56±0.02 | −55.20±2.06 | −55.20±2.06 | −55.20±2.06 | −55.20±2.06 |
| DEPF | −2.99±0.23 | −97.69±18.24 | −97.69±18.24 | −97.69±18.24 | −97.69±18.24 | −2.95±0.27 | −96.49±21.55 | −96.49±21.55 | −96.49±21.55 | −96.49±21.55 | −2.95±0.32 | −96.00±22.30 | −96.00±22.30 | −96.00±22.30 | −96.00±22.30 |
| DESPF | −2.96±0.21 | −94.92±18.74 | −94.92±18.74 | −94.92±18.74 | −94.92±18.74 | −2.92±0.26 | −93.09±22.05 | −93.09±22.05 | −93.09±22.05 | −93.09±22.05 | −2.92±0.32 | −93.08±21.24 | −93.08±21.24 | −93.08±21.24 | −93.08±21.24 |
| PRIME | −2.57±0.02 | −54.94±1.94 | −54.94±1.94 | −54.94±1.94 | −54.94±1.94 | −2.57±0.02 | −56.45±2.42 | −56.45±2.38 | −56.45±2.38 | −56.45±2.38 | −2.57±0.02 | −56.14±2.10 | −56.14±2.10 | −56.14±2.10 | −56.14±2.10 |
| Task | | | | | | | | CEC 4 | | | | | | | |
| $f(x^*_{DEPF})$ | | | | | | | | −264.33±0.01 | | | | | | | |
| CARCOO | | | | | | | | | | | | | | | |
| DEPF | −306.44±31.28 | −2.52e3.00±2.92e3.00 | −2.52e3.00±2.92e3.00 | −2.52e3.00±2.92e3.00 | −2.52e3.00±2.92e3.00 | −321.85±37.63 | −2.82e3.00±3.35e3.00 | −2.82e3.00±3.35e3.00 | −2.82e3.00±3.35e3.00 | −2.82e3.00±3.35e3.00 | −319.81±39.00 | −3.00e3.00±3.55e3.00 | −3.00e3.00±3.55e3.00 | −3.00e3.00±3.55e3.00 | −3.00e3.00±3.55e3.00 |
| DESPF | −382.83±0.03 | −1.17e4.00±3.29.66 | −1.17e4.00±3.29.66 | −1.17e4.00±3.29.66 | −1.17e4.00±3.29.66 | −382.83±0.03 | −1.18e4.00±3.27.46 | −1.18e4.00±3.27.46 | −1.18e4.00±3.27.46 | −1.18e4.00±3.27.46 | −382.83±0.03 | −1.18e4.00±3.27.66 | −1.18e4.00±3.27.66 | −1.18e4.00±3.27.66 | −1.18e4.00±3.27.66 |
| PRIME | −269.17±0.68 | −477.08±97.49 | −477.08±97.49 | −477.08±97.49 | −477.08±97.49 | −269.17±0.68 | −500.68±112.87 | −500.68±112.87 | −500.68±112.87 | −500.68±112.87 | −269.17±0.68 | −494.41±107.11 | −494.41±107.11 | −494.41±107.11 | −494.41±107.11 |
| Task | | | | | | | | CEC 5 | | | | | | | |
| $f(x^*_{DEPF})$ | | | | | | | | −2.95±0.08 | | | | | | | |
| CARCOO | −13.06±0.19 | −11.46±2.23 | −11.46±2.23 | −11.46±2.23 | −11.46±2.23 | −13.06±0.19 | −11.52±2.24 | −11.52±2.24 | −11.52±2.24 | −11.52±2.24 | −13.06±0.19 | −11.54±2.25 | −11.54±2.25 | −11.54±2.25 | −11.54±2.25 |
| DEPF | −29.74±10.36 | −2.22e3.00±1.26e3.00 | −2.22e3.00±1.26e3.00 | −2.22e3.00±1.26e3.00 | −2.22e3.00±1.26e3.00 | −38.16±13.62 | −2.38e3.00±1.38e3.00 | −2.38e3.00±1.38e3.00 | −2.38e3.00±1.38e3.00 | −2.38e3.00±1.38e3.00 | −67.28±6.00 | −2.83e3.00±1.55e3.00 | −2.83e3.00±1.55e3.00 | −2.83e3.00±1.55e3.00 | −2.83e3.00±1.55e3.00 |
| DESPF | −67.28±6.00 | −5.42e3.00±1.99e3.00 | −5.42e3.00±1.99e3.00 | −5.42e3.00±1.99e3.00 | −5.42e3.00±1.99e3.00 | −67.28±6.00 | −5.54e3.00±2.03e3.00 | −5.54e3.00±2.03e3.00 | −5.54e3.00±2.03e3.00 | −5.54e3.00±2.03e3.00 | −67.28±6.00 | −5.57e3.00±2.05e3.00 | −5.57e3.00±2.05e3.00 | −5.57e3.00±2.05e3.00 | −5.57e3.00±2.05e3.00 |
| PRIME | −13.06±0.19 | −508.84±489.87 | −508.84±489.87 | −508.84±489.87 | −508.84±489.87 | −13.06±0.19 | −511.50±492.42 | −511.50±492.42 | −511.50±492.42 | −511.50±492.42 | −13.06±0.19 | −512.37±493.26 | −512.37±493.26 | −512.37±493.26 | −512.37±493.26 |

Table 94: Overall results for GTOPX constrained tasks with 128 solutions and 100th percentile evaluations. In this case, 0% of the values are missing near the worst value and another 50% near the optimal value. Details are the same as Table 5.

| Steps | t − 50 | | | | | t − 100 | | | | | t − 150 | | | | |
|---|---|---|---|---|---|---|---|---|---|---|---|---|---|---|---|
| Task | | | | | | | | GTOPX 1 | | | | | | | |
| $f(x^*_{DEPF})$ | | | | | | | | −164.33±5.48 | | | | | | | |
| Metric | FS | SI | OI | SO | SO$_{ac}$ | FS | SI | OI | SO | SO$_{ac}$ | FS | SI | OI | SO | SO$_{ac}$ |
| CARCOO | | | | | | | | | | | | | | | |
| DEPF | −682.08±324.71 | −0.18±0.48 | −0.35±0.49 | −0.24±0.61 | −0.27±0.72 | −768.01±177.1 | −1.03±0.34 | −2.07±0.67 | −1.38±0.65 | −1.25±0.60 | −768.01±177.1 | −1.35±0.30 | −2.70±0.61 | −1.80±0.61 | −1.35±0.50 |
| DESPF | −768.01±177.1 | −3.17±5.00 | −6.33±5.86 | −4.22±5.77 | −4.76±3.90 | −768.01±177.1 | −3.36±5.90 | −6.71±5.79 | −4.47±5.51 | −4.04±6.28 | −768.01±177.1 | −3.42±5.49 | −6.84±5.75 | −4.56±5.52 | −3.43±5.60 |
| PRIME | | | | | | | | | | | | | | | |
| Task | | | | | | | | GTOPX 5 | | | | | | | |
| $f(x^*_{DEPF})$ | | | | | | | | 6.41±1.08 | | | | | | | |
| CARCOO | | | | | | | | | | | | | | | |
| DEPF | 0.00±0.00 | −632.95±187.60 | −632.95±187.40 | −632.95±187.60 | −632.95±187.40 | 0.00±0.00 | −636.25±188.38 | −636.25±188.38 | −636.25±188.38 | −636.25±188.38 | 0.00±0.00 | −637.33±188.70 | −637.33±188.70 | −637.33±188.70 | −637.33±188.70 |
| DESPF | 0.00±0.00 | −632.95±187.60 | −632.95±187.40 | −632.95±187.60 | −632.95±187.40 | 0.00±0.00 | −636.25±188.38 | −636.25±188.38 | −636.25±188.38 | −636.25±188.38 | 0.00±0.00 | −637.33±188.70 | −637.33±188.70 | −637.33±188.70 | −637.33±188.70 |
| PRIME | | | | | | | | | | | | | | | |
| Task | | | | | | | | GTOPX 7 | | | | | | | |
| $f(x^*_{DEPF})$ | | | | | | | | −670.34±5.15 | | | | | | | |
| CARCOO | −1.58e3.00±3.68e3.00 | 0.14±1.00 | 0.02±1.43 | 0.10±1.27 | 0.08±1.41 | −1.58e3.00±3.68e3.00 | 0.03±1.94 | −0.20±2.10 | −0.05±1.49 | −0.02±1.36 | −1.58e3.00±3.68e3.00 | −0.01±1.22 | −0.27±2.26 | −0.10±1.56 | −0.01±1.22 |
| DEPF | −4.37e3.00±481.68 | −1.70e5.00±1.75e5.00 | −1.70e5.00±1.75e5.00 | −1.70e5.00±1.75e5.00 | −1.70e5.00±1.75e5.00 | −4.37e3.00±481.68 | −1.71e5.00±1.76e5.00 | −1.71e5.00±1.76e5.00 | −1.71e5.00±1.76e5.00 | −1.71e5.00±1.76e5.00 | −4.37e3.00±481.68 | −1.72e5.00±1.76e5.00 | −1.72e5.00±1.76e5.00 | −1.72e5.00±1.76e5.00 | −1.72e5.00±1.76e5.00 |
| DESPF | −4.37e3.00±481.68 | −1.73e5.00±1.75e5.00 | −1.73e5.00±1.75e5.00 | −1.73e5.00±1.75e5.00 | −1.73e5.00±1.75e5.00 | −4.37e3.00±481.68 | −1.78e5.00±1.83e5.00 | −1.78e5.00±1.83e5.00 | −1.78e5.00±1.83e5.00 | −1.78e5.00±1.83e5.00 | −4.37e3.00±481.68 | −1.82e5.00±1.88e5.00 | −1.82e5.00±1.88e5.00 | −1.82e5.00±1.88e5.00 | −1.82e5.00±1.88e5.00 |
| PRIME | −4.32e3.00±496.57 | −2.84±4.18 | −5.67±6.76 | −3.78±5.94 | −4.26±6.59 | −4.32e3.00±496.57 | −4.10±5.24 | −8.20±10.48 | −5.47±6.99 | −4.94±6.30 | −4.32e3.00±496.57 | −4.52±5.53 | −9.04±11.06 | −6.02±7.38 | −4.53±5.55 |

Table 95: Overall results for GTOPX constrained tasks with 128 solutions and 100th percentile evaluations. In this case, 0% of the values are missing near the worst value and another 40% near the optimal value. Details are the same as Table 5.

| Steps | t − 50 | | | | | t − 100 | | | | | t − 150 | | | | |
|---|---|---|---|---|---|---|---|---|---|---|---|---|---|---|---|
| Task | | | | | | | | GTOPX 1 | | | | | | | |
| $f(x^*_{DEPF})$ | | | | | | | | −115.24±0.81 | | | | | | | |
| Metric | FS | SI | OI | SO | SO$_{ac}$ | FS | SI | OI | SO | SO$_{ac}$ | FS | SI | OI | SO | SO$_{ac}$ |
| CARCOO | −114.57±1.49 | 0.92±0.09 | 1.09±0.15 | 0.98±0.16 | 1.01±0.09 | −111.99±3.94 | 0.72±0.36 | 1.18±0.35 | 0.88±0.39 | 0.81±0.37 | −112.27±3.75 | 0.66±0.15 | 1.24±0.05 | 0.86±0.20 | 0.67±0.15 |
| DEPF | −682.75±214.42 | −3.59±2.64 | −4.79±3.54 | −0.63±4.98 | −0.71±3.49 | −768.01±177.71 | −1.81±0.36 | −3.62±1.11 | −2.41±0.76 | −2.17±0.67 | −768.01±177.71 | −2.45±0.63 | −4.91±0.86 | −3.27±0.67 | −2.46±0.63 |
| DESPF | −768.01±177.71 | −1.61e4.00±2.79e4.00 | −1.61e4.00±2.79e4.00 | −1.61e4.00±2.79e4.00 | −1.61e4.00±2.79e4.00 | −768.01±177.71 | −1.62e4.00±2.81e4.00 | −1.63e4.00±2.81e4.00 | −1.63e4.00±2.81e4.00 | −1.63e4.00±2.81e4.00 | −768.01±177.71 | −1.63e4.00±2.82e4.00 | −1.63e4.00±2.82e4.00 | −1.63e4.00±2.82e4.00 | −1.63e4.00±2.82e4.00 |
| PRIME | −768.01±177.71 | −4.90±2.60 | −9.80±4.87 | −6.53±3.27 | −7.36±3.66 | −768.01±177.71 | −5.76±2.90 | −11.52±5.79 | −7.68±3.86 | −6.93±3.68 | −768.01±177.71 | −6.04±3.05 | −12.08±6.10 | −8.05±4.07 | −6.06±3.06 |
| Task | | | | | | | | GTOPX 5 | | | | | | | |
| $f(x^*_{DEPF})$ | | | | | | | | 256.92±16.59 | | | | | | | |
| CARCOO | | | | | | | | | | | | | | | |
| DEPF | 0.00±0.00 | −2.54e4.00±3.13e3.00 | −2.54e4.00±3.13e3.00 | −2.54e4.00±3.13e3.00 | −2.54e4.00±3.13e3.00 | 0.00±0.00 | −2.56e4.00±3.14e3.00 | −2.56e4.00±3.14e3.00 | −2.56e4.00±3.14e3.00 | −2.56e4.00±3.14e3.00 | 0.00±0.00 | −2.56e4.00±3.15e3.00 | −2.56e4.00±3.15e3.00 | −2.56e4.00±3.15e3.00 | −2.56e4.00±3.15e3.00 |
| DESPF | 0.00±0.00 | −2.54e4.00±3.13e3.00 | −2.54e4.00±3.13e3.00 | −2.54e4.00±3.13e3.00 | −2.54e4.00±3.13e3.00 | 0.00±0.00 | −2.56e4.00±3.14e3.00 | −2.56e4.00±3.14e3.00 | −2.56e4.00±3.14e3.00 | −2.56e4.00±3.14e3.00 | 0.00±0.00 | −2.56e4.00±3.15e3.00 | −2.56e4.00±3.15e3.00 | −2.56e4.00±3.15e3.00 | −2.56e4.00±3.15e3.00 |
| PRIME | | | | | | | | | | | | | | | |
| Task | | | | | | | | GTOPX 7 | | | | | | | |
| $f(x^*_{DEPF})$ | | | | | | | | −568.34±5.15 | | | | | | | |
| CARCOO | −1.03e3.00±3.68e3.00 | 0.24±3.12 | 0.36±2.18 | 0.28±1.47 | 0.30±3.65 | −1.03e3.00±3.68e3.00 | 0.18±3.28 | 0.24±2.60 | 0.20±1.68 | 0.20±1.68 | −1.03e3.00±3.68e3.00 | 0.16±3.28 | 0.21±2.51 | 0.18±1.69 | 0.16±1.29 |
| DEPF | −4.37e3.00±481.68 | −1.05e5.00±1.82e5.00 | −1.05e5.00±1.82e5.00 | −1.05e5.00±1.82e5.00 | −1.05e5.00±1.82e5.00 | −4.37e3.00±481.68 | −1.06e5.00±1.82e5.00 | −1.06e5.00±1.82e5.00 | −1.06e5.00±1.82e5.00 | −1.06e5.00±1.82e5.00 | −4.37e3.00±481.68 | −1.06e5.00±1.84e5.00 | −1.06e5.00±1.84e5.00 | −1.06e5.00±1.84e5.00 | −1.06e5.00±1.84e5.00 |
| DESPF | −4.37e3.00±481.68 | −2.40e5.00±3.38e5.00 | −2.40e5.00±3.38e5.00 | −2.40e5.00±3.38e5.00 | −2.40e5.00±3.38e5.00 | −4.37e3.00±481.68 | −2.45e5.00±3.48e5.00 | −2.45e5.00±3.48e5.00 | −2.45e5.00±3.48e5.00 | −2.45e5.00±3.48e5.00 | −4.37e3.00±481.68 | −2.46e5.00±3.49e5.00 | −2.46e5.00±3.49e5.00 | −2.46e5.00±3.49e5.00 | −2.46e5.00±3.49e5.00 |
| PRIME | −4.37e3.00±481.68 | −1.32±0.68 | −2.63±1.21 | −1.75±0.81 | −1.98±0.91 | −4.37e3.00±481.68 | −2.23±0.46 | −4.46±0.92 | −2.97±0.62 | −2.68±0.58 | −4.37e3.00±481.68 | −2.53±0.45 | −5.06±0.90 | −3.37±0.60 | −2.54±0.45 |

Table 96: Overall results for GTOPX unconstrained tasks with 128 solutions and 100th percentile evaluations. In this case, 0% of the values are missing near the worst value and another 30% near the optimal value. Details are the same as Table 5.

| Steps | t − 50 | | | | | t − 100 | | | | | t − 150 | | | | |
|---|---|---|---|---|---|---|---|---|---|---|---|---|---|---|---|
| Task | | | | | | | | GTOPX 1 | | | | | | | |
| $f(x^*_{DEPF})$ | | | | | | | | −86.91±0.50 | | | | | | | |
| Metric | FS | SI | OI | SO | SO$_{ac}$ | FS | SI | OI | SO | SO$_{ac}$ | FS | SI | OI | SO | SO$_{ac}$ |
| CARCOO | −85.01±4.17 | 0.93±0.09 | 1.09±0.14 | 0.99±0.001 | 1.02±0.05 | −86.18±3.94 | 0.91±0.13 | 1.11±0.16 | 0.98±0.04 | 0.96±0.06 | −85.63±2.09 | 0.88±0.16 | 1.14±0.19 | 0.97±0.06 | 0.88±0.13 |
| DEPF | −682.75±214.42 | −3.59±2.64 | −7.18±5.56 | −4.79±3.54 | −5.39±3.98 | −676.65±263.33 | −5.55±2.90 | −11.10±5.79 | −7.40±3.86 | −6.68±3.68 | −768.01±177.71 | −6.25±3.07 | −12.49±6.13 | −8.33±4.09 | −6.27±3.08 |
| DESPF | −768.01±177.71 | −1.66e4.00±2.88e4.00 | −1.66e4.00±2.88e4.00 | −1.66e4.00±2.88e4.00 | −1.66e4.00±2.88e4.00 | −768.01±177.71 | −1.68e4.00±2.95e4.00 | −1.68e4.00±2.95e4.00 | −1.68e4.00±2.95e4.00 | −1.68e4.00±2.95e4.00 | −768.01±177.71 | −1.68e4.00±2.95e4.00 | −1.68e4.00±2.95e4.00 | −1.68e4.00±2.95e4.00 | −1.68e4.00±2.95e4.00 |
| PRIME | −768.01±177.71 | −3.09±0.99 | −6.17±1.98 | −4.11±1.32 | −4.64±1.49 | −768.01±177.71 | −5.00±1.60 | −10.01±3.27 | −6.67±2.18 | −6.02±2.19 | −768.01±177.71 | −5.64±1.80 | −11.27±3.16 | −7.51±1.80 | −5.65±1.08 |
| Task | | | | | | | | GTOPX 5 | | | | | | | |
| $f(x^*_{DEPF})$ | | | | | | | | 2.20e3±731.08 | | | | | | | |
| CARCOO | 4.12e4.00±4.74e4.00 | 0.82±0.06 | 1.32±0.12 | 1.01±0.09 | 1.09±0.08 | 5.39e4.00±5.68e4.00 | 0.76±0.04 | 1.45±0.09 | 0.99±0.09 | 0.90±0.60 | 4.49e4.00±5.38e4.00 | 0.75±0.09 | 1.50±0.08 | 1.00±0.08 | 0.75±0.01 |
| DEPF | 0.00±0.00 | −2.17e6.00±3.78e6.00 | −2.17e6.00±3.78e6.00 | −2.17e6.00±3.78e6.00 | −2.17e6.00±3.78e6.00 | 0.00±0.00 | −2.19e6.00±3.78e6.00 | −2.19e6.00±3.78e6.00 | −2.19e6.00±3.78e6.00 | −2.19e6.00±3.78e6.00 | 0.00±0.00 | −2.19e6.00±3.73e6.00 | −2.19e6.00±3.73e6.00 | −2.19e6.00±3.73e6.00 | −2.19e6.00±3.73e6.00 |
| DESPF | 0.00±0.00 | −2.17e6.00±3.78e6.00 | −2.17e6.00±3.78e6.00 | −2.17e6.00±3.78e6.00 | −2.17e6.00±3.78e6.00 | 0.00±0.00 | −2.19e6.00±3.78e6.00 | −2.19e6.00±3.78e6.00 | −2.19e6.00±3.78e6.00 | −2.19e6.00±3.78e6.00 | 0.00±0.00 | −2.19e6.00±3.73e6.00 | −2.19e6.00±3.73e6.00 | −2.19e6.00±3.73e6.00 | −2.19e6.00±3.73e6.00 |
| PRIME | 0.00±0.00 | 0.25±0.12 | 0.51±0.25 | 0.34±0.17 | 0.00±0.00 | 0.00±0.00 | 0.23±0.13 | 0.46±0.26 | 0.30±0.17 | 0.28±0.16 | 0.00±0.00 | 0.22±0.13 | 0.44±0.26 | 0.29±0.17 | 0.22±0.13 |
| Task | | | | | | | | GTOPX 7 | | | | | | | |
| $f(x^*_{DEPF})$ | | | | | | | | −423.43±5.15 | | | | | | | |
| CARCOO | −351.83±115.46 | 0.68±0.15 | 1.24±0.30 | 0.87±0.12 | 0.96±0.15 | −399.50±66.47 | 0.64±0.15 | 1.16±0.17 | 0.82±0.12 | 0.75±0.13 | −399.50±66.47 | 0.63±0.16 | 1.13±0.16 | 0.79±0.13 | 0.63±0.16 |
| DEPF | −3.88e3.00±3.52e3.00 | −2.88e5.00±3.73e5.00 | −2.88e5.00±3.73e5.00 | −2.88e5.00±3.73e5.00 | −2.88e5.00±3.73e5.00 | −4.05e3.00±481.68 | −2.91e5.00±3.69e5.00 | −2.91e5.00±3.69e5.00 | −2.91e5.00±3.69e5.00 | −2.91e5.00±3.69e5.00 | −4.05e3.00±481.68 | −2.93e5.00±3.71e5.00 | −2.93e5.00±3.71e5.00 | −2.93e5.00±3.71e5.00 | −2.93e5.00±3.71e5.00 |
| DESPF | −4.37e3.00±481.68 | −2.63e5.00±3.65e5.00 | −2.63e5.00±3.65e5.00 | −2.63e5.00±3.65e5.00 | −2.63e5.00±3.65e5.00 | −4.37e3.00±481.68 | −2.78e5.00±3.69e5.00 | −2.78e5.00±3.69e5.00 | −2.78e5.00±3.69e5.00 | −2.78e5.00±3.69e5.00 | −4.37e3.00±481.68 | −2.83e5.00±3.71e5.00 | −2.83e5.00±3.71e5.00 | −2.83e5.00±3.71e5.00 | −2.83e5.00±3.71e5.00 |
| PRIME | −4.37e3.00±481.68 | −3.10±0.58 | −6.20±1.48 | −4.14±0.88 | −4.66±1.11 | −4.37e3.00±481.68 | −4.03±0.73 | −8.07±1.45 | −5.38±0.97 | −4.85±0.98 | −4.37e3.00±481.68 | −4.34±0.73 | −8.68±1.46 | −5.79±0.97 | −4.36±0.73 |

Table 97: Overall results for GTOPX constrained tasks with 128 solutions and 100th percentile evaluations. In this case, 0% of the values are missing near the worst value and another 20% near the optimal value. Details are the same as Table 5.

| Steps | | t − 50 | | | | | t − 100 | | | | | t − 150 | | | |
|---|---|---|---|---|---|---|---|---|---|---|---|---|---|---|---|
| Task | | | | | | | | GTOPX 1 | | | | | | | |
| $f(x^*_{DEF})$ | | | | | | | | -64.89 ±0.61 | | | | | | | |
| Metric | FS | SI | OI | SO | $SO_c$ | FS | SI | OI | SO | $SO_c$ | FS | SI | OI | SO | $SO_c$ |
| CARCOO | -63.91 ±2.16 | 0.91 ±0.14 | 1.09 ±0.15 | 0.98 ±0.11 | 1.01 ±0.10 | -62.56 ±2.53 | 0.75 ±0.16 | 1.19 ±0.11 | 0.91 ±0.13 | 0.85 ±0.14 | -60.59 ±4.87 | 0.74 ±0.16 | 1.27 ±0.15 | 0.92 ±0.12 | 0.74 ±0.16 |
| DEPF | -611.17 ±275.60 | -4.73 ±2.66 | -9.45 ±5.30 | -6.30 ±3.91 | -7.10 ±4.00 | -768.01 ±17.71 | -9.11 ±5.12 | -18.21 ±6.29 | -12.14 ±6.56 | -10.96 ±5.74 | -768.01 ±17.71 | -10.93 ±5.57 | -21.86 ±7.14 | -14.58 ±6.51 | -10.97 ±5.58 |
| DESPF | -768.01 ±17.71 | -4.22e4.00 ±1.36e4.00 | -4.22e4.00 ±1.36e4.00 | -4.22e4.00 ±1.36e4.00 | -4.22e4.00 ±1.36e4.00 | -768.01 ±17.71 | -4.28e4.00 ±1.51e4.00 | -4.28e4.00 ±1.51e4.00 | -4.28e4.00 ±1.51e4.00 | -4.28e4.00 ±1.51e4.00 | -768.01 ±17.71 | -4.30e4.00 ±1.63e4.00 | -4.30e4.00 ±1.63e4.00 | -4.30e4.00 ±1.63e4.00 | -4.30e4.00 ±1.63e4.00 |
| PRIME | -768.01 ±17.71 | -5.14 ±1.52 | -10.27 ±3.80 | -6.85 ±2.02 | -7.72 ±2.28 | -768.01 ±17.71 | -7.77 ±1.82 | -15.54 ±2.44 | -10.36 ±2.43 | -9.35 ±2.19 | -768.01 ±17.71 | -8.63 ±1.95 | -17.27 ±3.390 | -11.51 ±2.40 | -8.66 ±1.96 |
| Task | | | | | | | | GTOPX 5 | | | | | | | |
| $f(x^*_{DEF})$ | | | | | | | | 5.40e4 ±2.08e3.00 | | | | | | | |
| CARCOO | 1.07e5.00 ±3.16e4.00 | 0.77 ±0.04 | 1.51 ±0.08 | 1.02 ±0.03 | 1.14 ±0.06 | 6.84e4.00 ±1.95e4.00 | 0.69 ±0.05 | 1.39 ±0.11 | 0.93 ±0.07 | 0.83 ±0.06 | 6.51e4.00 ±3.50e4.00 | 0.65 ±0.05 | 1.30 ±0.11 | 0.87 ±0.07 | 0.65 ±0.06 |
| DEPF | 0.00 ±0.00 | -5.34e6.00 ±1.98e5.00 | -5.34e6.00 ±1.98e5.00 | -5.34e6.00 ±1.98e5.00 | -5.34e6.00 ±1.98e5.00 | 0.00 ±0.00 | -5.37e6.00 ±1.99e5.00 | -5.37e6.00 ±1.99e5.00 | -5.37e6.00 ±1.99e5.00 | -5.37e6.00 ±1.99e5.00 | 0.00 ±0.00 | -5.38e6.00 ±1.99e5.00 | -5.38e6.00 ±1.99e5.00 | -5.38e6.00 ±1.99e5.00 | -5.38e6.00 ±1.99e5.00 |
| DESPF | 0.00 ±0.00 | -5.34e6.00 ±1.98e5.00 | -5.34e6.00 ±1.98e5.00 | -5.34e6.00 ±1.98e5.00 | -5.34e6.00 ±1.98e5.00 | 0.00 ±0.00 | -5.37e6.00 ±1.99e5.00 | -5.37e6.00 ±1.99e5.00 | -5.37e6.00 ±1.99e5.00 | -5.37e6.00 ±1.99e5.00 | 0.00 ±0.00 | -5.38e6.00 ±1.99e5.00 | -5.38e6.00 ±1.99e5.00 | -5.38e6.00 ±1.99e5.00 | -5.38e6.00 ±1.99e5.00 |
| PRIME | 0.00 ±0.00 | 0.17 ±0.09 | 0.33 ±0.17 | 0.22 ±0.05 | 0.25 ±0.38 | 0.00 ±0.00 | 0.14 ±0.19 | 0.28 ±0.16 | 0.19 ±0.26 | 0.17 ±0.23 | 0.00 ±0.00 | 0.13 ±0.19 | 0.27 ±0.39 | 0.18 ±0.26 | 0.13 ±0.20 |
| Task | | | | | | | | GTOPX 7 | | | | | | | |
| $f(x^*_{DEF})$ | | | | | | | | -277.23 ±4.46 | | | | | | | |
| CARCOO | -279.55 ±4.64 | 0.60 ±0.15 | 1.07 ±0.04 | 0.75 ±0.09 | 0.83 ±0.07 | -279.55 ±4.64 | 0.58 ±0.16 | 1.03 ±0.02 | 0.73 ±0.08 | 0.67 ±0.12 | -279.55 ±4.64 | 0.57 ±0.16 | 1.01 ±0.02 | 0.72 ±0.11 | 0.57 ±0.16 |
| DEPF | -403e3.00 ±1.17e3.00 | -3.06e5.00 ±1.40e5.00 | -3.06e5.00 ±1.40e5.00 | -3.06e5.00 ±1.40e5.00 | -3.06e5.00 ±1.40e5.00 | -4.37e3.00 ±481.60 | -3.34e5.00 ±1.35e5.00 | -3.34e5.00 ±1.45e5.00 | -3.34e5.00 ±1.45e5.00 | -3.34e5.00 ±1.45e5.00 | -4.37e3.00 ±481.60 | -3.45e5.00 ±1.64e5.00 | -3.45e5.00 ±1.64e5.00 | -3.45e5.00 ±1.64e5.00 | -3.45e5.00 ±1.64e5.00 |
| DESPF | -4.37e3.00 ±481.60 | -2.96e5.00 ±1.37e5.00 | -2.96e5.00 ±1.37e5.00 | -2.96e5.00 ±1.37e5.00 | -2.96e5.00 ±1.37e5.00 | -4.37e3.00 ±481.60 | -2.94e5.00 ±1.74e5.00 | -2.94e5.00 ±1.74e5.00 | -2.94e5.00 ±1.74e5.00 | -2.94e5.00 ±1.74e5.00 | -4.37e3.00 ±481.60 | -2.99e5.00 ±1.75e5.00 | -2.99e5.00 ±1.75e5.00 | -2.99e5.00 ±1.75e5.00 | -2.99e5.00 ±1.75e5.00 |
| PRIME | -4.31e3.00 ±484.54 | -57.81 ±114.89 | -115.63 ±229.74 | -77.08 ±153.18 | -86.86 ±172.62 | -4.31e3.00 ±484.54 | -62.78 ±123.61 | -125.57 ±247.21 | -83.71 ±166.31 | -75.54 ±148.72 | -4.31e3.00 ±484.54 | -64.42 ±126.37 | -128.84 ±252.96 | -85.89 ±168.63 | -64.63 ±126.09 |

Table 98: Overall results for GTOPX constrained tasks with 128 solutions and 100th percentile evaluations. In this case, 0% of the values are missing near the worst value and another 10% near the optimal value. Details are the same as Table 5.

| Steps | | t − 50 | | | | | t − 100 | | | | | t − 150 | | | |
|---|---|---|---|---|---|---|---|---|---|---|---|---|---|---|---|
| Task | | | | | | | | GTOPX 1 | | | | | | | |
| $f(x^*_{DEF})$ | | | | | | | | -47.69 ±0.61 | | | | | | | |
| Metric | FS | SI | OI | SO | $SO_c$ | FS | SI | OI | SO | $SO_c$ | FS | SI | OI | SO | $SO_c$ |
| CARCOO | -47.69 ±0.92 | 0.70 ±0.37 | 0.78 ±0.38 | 0.73 ±0.37 | 0.74 ±0.37 | -47.68 ±0.90 | 0.67 ±0.38 | 0.79 ±0.42 | 0.71 ±0.39 | 0.70 ±0.59 | -47.67 ±0.89 | 0.65 ±0.39 | 0.81 ±0.45 | 0.67 ±0.39 | 0.60 ±0.59 |
| DEPF | -612.43 ±374.12 | -7.25e3.00 ±3.06e4.00 | -7.30e3.00 ±3.19e4.00 | -7.27e3.00 ±3.98e4.00 | -7.28e3.00 ±3.90e4.00 | -768.01 ±17.71 | -8.31e3.00 ±3.34e4.00 | -8.37e3.00 ±2.95e4.00 | -8.33e3.00 ±2.59e4.00 | -8.32e3.00 ±2.78e4.00 | -768.01 ±17.71 | -8.66e3.00 ±3.25e4.00 | -8.73e3.00 ±2.75e4.00 | -8.68e3.00 ±2.25e4.00 | -8.66e3.00 ±2.25e4.00 |
| DESPF | -768.01 ±17.71 | -5.20e4.00 ±3.08e4.00 | -5.20e4.00 ±3.08e4.00 | -5.20e4.00 ±3.08e4.00 | -5.20e4.00 ±3.08e4.00 | -768.01 ±17.71 | -5.32e4.00 ±3.07e4.00 | -5.32e4.00 ±3.07e4.00 | -5.32e4.00 ±3.07e4.00 | -5.32e4.00 ±3.07e4.00 | -768.01 ±17.71 | -5.36e4.00 ±3.09e4.00 | -5.36e4.00 ±3.08e4.00 | -5.36e4.00 ±3.08e4.00 | -5.36e4.00 ±3.09e4.00 |
| PRIME | -768.01 ±17.71 | -8.83 ±4.88 | -17.65 ±8.03 | -11.77 ±5.35 | -13.26 ±6.05 | -768.01 ±17.71 | -12.60 ±5.19 | -25.20 ±6.67 | -16.80 ±4.25 | -15.16 ±3.83 | -768.01 ±17.71 | -13.84 ±5.00 | -27.69 ±6.35 | -18.46 ±4.12 | -13.89 ±3.10 |
| Task | | | | | | | | GTOPX 5 | | | | | | | |
| $f(x^*_{DEF})$ | | | | | | | | 1.10e5 ±3.54e3.00 | | | | | | | |
| CARCOO | 1.26e5.00 ±2.97e4.00 | 0.66 ±0.08 | 1.32 ±0.16 | 0.88 ±0.11 | 0.99 ±0.12 | 1.10e5.00 ±3.76e4.00 | 0.58 ±0.03 | 1.16 ±0.09 | 0.78 ±0.06 | 0.70 ±0.06 | 1.10e5.00 ±3.50e4.00 | 0.55 ±0.03 | 1.11 ±0.06 | 0.74 ±0.08 | 0.56 ±0.03 |
| DEPF | 0.00 ±0.00 | -1.09e7.00 ±3.53e5.00 | -1.09e7.00 ±3.53e5.00 | -1.09e7.00 ±3.53e5.00 | -1.09e7.00 ±3.53e5.00 | 0.00 ±0.00 | -1.10e7.00 ±3.53e5.00 | -1.10e7.00 ±3.53e5.00 | -1.10e7.00 ±3.53e5.00 | -1.10e7.00 ±3.53e5.00 | 0.00 ±0.00 | -1.10e7.00 ±3.53e5.00 | -1.10e7.00 ±3.53e5.00 | -1.10e7.00 ±3.53e5.00 | -1.10e7.00 ±3.53e5.00 |
| DESPF | 0.00 ±0.00 | -1.09e7.00 ±3.53e5.00 | -1.09e7.00 ±3.53e5.00 | -1.09e7.00 ±3.53e5.00 | -1.09e7.00 ±3.53e5.00 | 0.00 ±0.00 | -1.10e7.00 ±3.53e5.00 | -1.10e7.00 ±3.53e5.00 | -1.10e7.00 ±3.53e5.00 | -1.10e7.00 ±3.53e5.00 | 0.00 ±0.00 | -1.10e7.00 ±3.53e5.00 | -1.10e7.00 ±3.53e5.00 | -1.10e7.00 ±3.53e5.00 | -1.10e7.00 ±3.53e5.00 |
| PRIME | 0.00 ±0.00 | -0.03 ±0.17 | -0.07 ±0.75 | -0.04 ±0.50 | -0.05 ±0.56 | 0.00 ±0.00 | -0.06 ±0.10 | -0.12 ±0.76 | -0.08 ±0.51 | -0.07 ±0.66 | 0.00 ±0.00 | -0.07 ±0.38 | -0.14 ±0.77 | -0.09 ±0.51 | -0.07 ±0.38 |
| Task | | | | | | | | GTOPX 7 | | | | | | | |
| $f(x^*_{DEF})$ | | | | | | | | -149.12 ±2.17 | | | | | | | |
| CARCOO | -151.22 ±3.32 | 0.75 ±0.26 | 1.01 ±0.02 | 0.84 ±0.16 | 0.88 ±0.12 | -151.22 ±3.32 | 0.75 ±0.25 | 0.99 ±0.02 | 0.83 ±0.17 | 0.80 ±0.28 | -151.22 ±3.32 | 0.74 ±0.26 | 0.99 ±0.02 | 0.82 ±0.17 | 0.74 ±0.26 |
| DEPF | -4.37e3.00 ±481.60 | -2.85e5.00 ±1.73e5.00 | -2.85e5.00 ±1.73e5.00 | -2.85e5.00 ±1.73e5.00 | -2.85e5.00 ±1.73e5.00 | -4.37e3.00 ±481.60 | -3.01e5.00 ±1.48e5.00 | -3.01e5.00 ±1.86e5.00 | -3.01e5.00 ±1.86e5.00 | -3.01e5.00 ±1.86e5.00 | -4.37e3.00 ±481.60 | -3.06e5.00 ±1.82e5.00 | -3.06e5.00 ±1.82e5.00 | -3.06e5.00 ±1.82e5.00 | -3.06e5.00 ±1.82e5.00 |
| DESPF | -4.37e3.00 ±481.60 | -3.53e5.00 ±1.18e5.00 | -3.53e5.00 ±1.18e5.00 | -3.53e5.00 ±1.18e5.00 | -3.53e5.00 ±1.18e5.00 | -4.37e3.00 ±481.60 | -3.64e5.00 ±1.30e5.00 | -3.64e5.00 ±1.30e5.00 | -3.64e5.00 ±1.30e5.00 | -3.64e5.00 ±1.30e5.00 | -4.37e3.00 ±481.60 | -3.68e5.00 ±1.38e5.00 | -3.68e5.00 ±1.38e5.00 | -3.68e5.00 ±1.38e5.00 | -3.68e5.00 ±1.38e5.00 |
| PRIME | -4.37e3.00 ±481.60 | -30.78 ±19.99 | -61.56 ±39.98 | -41.04 ±26.65 | -46.25 ±30.03 | -4.37e3.00 ±481.60 | -34.05 ±21.51 | -68.10 ±43.09 | -45.40 ±28.72 | -40.97 ±25.92 | -4.37e3.00 ±481.60 | -35.12 ±32.06 | -70.25 ±64.12 | -46.83 ±25.05 | -35.24 ±22.13 |

Table 99: Overall results for GTOPX constrained tasks with 128 solutions and 100th percentile evaluations. In this case, 10% of the values are missing near the worst value and another 40% near the optimal value. Details are the same as Table 5.

| Steps | | t − 50 | | | | | t − 100 | | | | | t − 150 | | | |
|---|---|---|---|---|---|---|---|---|---|---|---|---|---|---|---|
| Task | | | | | | | | GTOPX 1 | | | | | | | |
| $f(x^*_{DEF})$ | | | | | | | | -115.24 ±0.63 | | | | | | | |
| Metric | FS | SI | OI | SO | $SO_c$ | FS | SI | OI | SO | $SO_c$ | FS | SI | OI | SO | $SO_c$ |
| CARCOO | -114.70 ±1.17 | 0.71 ±0.54 | 0.77 ±0.56 | 0.74 ±0.55 | 0.75 ±0.55 | -113.28 ±3.63 | 0.60 ±0.52 | 0.83 ±0.61 | 0.69 ±0.55 | 0.57 ±0.52 | -112.29 ±1.16 | 0.57 ±0.52 | 0.89 ±0.65 | 0.68 ±0.55 | 0.57 ±0.52 |
| DEPF | -263.29 ±210.69 | -0.19 ±0.64 | -0.39 ±1.25 | -0.26 ±0.84 | -0.29 ±0.94 | -530.54 ±5.24 | -1.00 ±2.38 | -2.01 ±0.77 | -1.34 ±0.51 | -1.21 ±0.46 | -530.54 ±5.24 | -1.43 ±0.27 | -2.86 ±0.54 | -1.91 ±0.36 | -1.43 ±0.27 |
| DESPF | -530.54 ±5.24 | -2.04e4.00 ±2.06e4.00 | -2.04e4.00 ±2.06e4.00 | -2.04e4.00 ±2.06e4.00 | -2.04e4.00 ±2.06e4.00 | -530.54 ±5.24 | -2.05e4.00 ±2.09e4.00 | -2.05e4.00 ±2.09e4.00 | -2.05e4.00 ±2.09e4.00 | -2.05e4.00 ±2.09e4.00 | -530.54 ±5.24 | -2.05e4.00 ±2.05e4.00 | -2.05e4.00 ±2.05e4.00 | -2.05e4.00 ±2.05e4.00 | -2.05e4.00 ±2.05e4.00 |
| PRIME | -530.54 ±5.24 | -2.19 ±0.94 | -4.37 ±1.87 | -2.91 ±1.25 | -3.28 ±1.41 | -530.54 ±5.24 | -2.64 ±1.05 | -5.27 ±1.05 | -3.52 ±1.37 | -3.17 ±1.20 | -530.54 ±5.24 | -2.79 ±1.06 | -5.57 ±2.12 | -3.71 ±1.41 | -2.79 ±1.06 |
| Task | | | | | | | | GTOPX 5 | | | | | | | |
| $f(x^*_{DEF})$ | | | | | | | | 256.92 ±31.59 | | | | | | | |
| CARCOO | - | - | - | - | - | - | - | - | - | - | - | - | - | - | - |
| DEPF | 0.00 ±0.00 | -2.54e4.00 ±3.13e3.00 | -2.54e4.00 ±3.13e3.00 | -2.54e4.00 ±3.13e3.00 | -2.54e4.00 ±3.13e3.00 | 0.00 ±0.00 | -2.56e4.00 ±3.13e3.00 | -2.56e4.00 ±3.13e3.00 | -2.56e4.00 ±3.13e3.00 | -2.56e4.00 ±3.13e3.00 | 0.00 ±0.00 | -2.56e4.00 ±3.15e3.00 | -2.56e4.00 ±3.15e3.00 | -2.56e4.00 ±3.15e3.00 | -2.56e4.00 ±3.15e3.00 |
| DESPF | 0.00 ±0.00 | -2.54e4.00 ±3.13e3.00 | -2.54e4.00 ±3.13e3.00 | -2.54e4.00 ±3.13e3.00 | -2.54e4.00 ±3.13e3.00 | 0.00 ±0.00 | -2.56e4.00 ±3.13e3.00 | -2.56e4.00 ±3.13e3.00 | -2.56e4.00 ±3.13e3.00 | -2.56e4.00 ±3.13e3.00 | 0.00 ±0.00 | -2.56e4.00 ±3.15e3.00 | -2.56e4.00 ±3.15e3.00 | -2.56e4.00 ±3.15e3.00 | -2.56e4.00 ±3.15e3.00 |
| PRIME | - | - | - | - | - | - | - | - | - | - | - | - | - | - | - |
| Task | | | | | | | | GTOPX 7 | | | | | | | |
| $f(x^*_{DEF})$ | | | | | | | | -568.34 ±3.01 | | | | | | | |
| CARCOO | -901.72 ±491.39 | 0.48 ±0.34 | 0.83 ±0.57 | 0.60 ±0.41 | 0.66 ±0.44 | -967.76 ±386.89 | 0.35 ±0.41 | 0.57 ±0.64 | 0.42 ±0.48 | 0.39 ±0.45 | -967.76 ±386.89 | 0.29 ±0.42 | 0.47 ±0.65 | 0.35 ±0.50 | 0.30 ±0.42 |
| DEPF | -1.37e3.00 ±9.85 | -4.68e4.00 ±3.23e4.00 | -4.68e4.00 ±3.23e4.00 | -4.68e4.00 ±3.23e4.00 | -4.68e4.00 ±3.23e4.00 | -1.29e3.00 ±399.10 | -5.30e4.00 ±3.57e4.00 | -5.30e4.00 ±3.57e4.00 | -5.30e4.00 ±3.57e4.00 | -5.30e4.00 ±3.57e4.00 | -1.37e3.00 ±9.85 | -5.52e4.00 ±3.12e4.00 | -5.52e4.00 ±3.12e4.00 | -5.52e4.00 ±3.12e4.00 | -5.52e4.00 ±3.12e4.00 |
| DESPF | -1.37e3.00 ±9.85 | -5.85e4.00 ±3.38e4.00 | -5.85e4.00 ±3.38e4.00 | -5.85e4.00 ±3.38e4.00 | -5.85e4.00 ±3.38e4.00 | -1.37e3.00 ±9.85 | -5.93e4.00 ±3.34e4.00 | -5.93e4.00 ±3.34e4.00 | -5.93e4.00 ±3.34e4.00 | -5.93e4.00 ±3.34e4.00 | -1.37e3.00 ±9.85 | -5.96e4.00 ±3.33e4.00 | -5.96e4.00 ±3.33e4.00 | -5.96e4.00 ±3.33e4.00 | -5.96e4.00 ±3.33e4.00 |
| PRIME | -1.37e3.00 ±9.85 | 0.11 ±0.30 | 0.22 ±0.59 | 0.15 ±0.39 | 0.17 ±0.43 | -1.37e3.00 ±9.85 | -0.19 ±0.46 | -0.39 ±0.92 | -0.26 ±0.61 | -0.23 ±0.55 | -1.37e3.00 ±9.85 | -0.29 ±0.32 | -0.58 ±1.03 | -0.39 ±0.69 | -0.29 ±0.52 |

Table 100: Overall results for GTOPX constrained tasks with 128 solutions and 100th percentile evaluations. In this case, 20% of the values are missing near the worst value and another 30% near the optimal value. Details are the same as Table 5.

| Steps | | t − 50 | | | | | t − 100 | | | | | t − 150 | | | |
|---|---|---|---|---|---|---|---|---|---|---|---|---|---|---|---|
| Task | | | | | | | | GTOPX 1 | | | | | | | |
| $f(x^*_{DEF})$ | | | | | | | | -86.91 ±0.70 | | | | | | | |
| Metric | FS | SI | OI | SO | $SO_c$ | FS | SI | OI | SO | $SO_c$ | FS | SI | OI | SO | $SO_c$ |
| CARCOO | -84.93 ±4.59 | 0.94 ±0.08 | 1.07 ±0.15 | 0.99 ±0.04 | 1.01 ±0.06 | -84.79 ±5.86 | 0.83 ±0.18 | 1.13 ±0.16 | 0.93 ±0.16 | 0.89 ±0.13 | -84.80 ±4.75 | 0.78 ±0.16 | 1.17 ±0.17 | 0.91 ±0.10 | 0.78 ±0.16 |
| DEPF | -346.54 ±116.33 | -6.76 ±3.68 | -13.53 ±27.62 | -9.02 ±18.42 | -10.16 ±20.73 | -389.73 ±11.96 | -7.79 ±14.47 | -15.58 ±28.95 | -10.39 ±19.30 | -9.37 ±17.41 | -389.73 ±11.96 | -8.23 ±14.66 | -16.47 ±29.26 | -10.98 ±19.50 | -8.26 ±14.68 |
| DESPF | -389.73 ±11.96 | -1.09e4.00 ±1.41e4.00 | -1.09e4.00 ±1.41e4.00 | -1.09e4.00 ±1.41e4.00 | -1.09e4.00 ±1.41e4.00 | -389.73 ±11.96 | -1.10e4.00 ±1.42e4.00 | -1.10e4.00 ±1.42e4.00 | -1.10e4.00 ±1.42e4.00 | -1.10e4.00 ±1.42e4.00 | -389.73 ±11.96 | -1.10e4.00 ±1.42e4.00 | -1.10e4.00 ±1.42e4.00 | -1.10e4.00 ±1.42e4.00 | -1.10e4.00 ±1.42e4.00 |
| PRIME | -389.73 ±11.96 | -0.97 ±0.40 | -1.95 ±0.80 | -1.30 ±0.53 | -1.46 ±0.60 | -389.73 ±11.96 | -1.65 ±0.47 | -3.30 ±0.73 | -2.20 ±0.49 | -1.99 ±0.44 | -389.73 ±11.96 | -1.87 ±0.37 | -3.75 ±0.73 | -2.50 ±0.49 | -1.88 ±0.37 |
| Task | | | | | | | | GTOPX 5 | | | | | | | |
| $f(x^*_{DEF})$ | | | | | | | | 2.20e4 ±733.38 | | | | | | | |
| CARCOO | 4.51e4.00 ±3.19e4.00 | 0.82 ±0.07 | 1.26 ±0.09 | 0.99 ±0.05 | 1.06 ±0.05 | 4.78e4.00 ±1.76e4.00 | 0.73 ±0.03 | 1.41 ±0.06 | 0.96 ±0.03 | 0.87 ±0.05 | 5.27e4.00 ±1.69e4.00 | 0.73 ±0.03 | 1.46 ±0.06 | 0.97 ±0.04 | 0.73 ±0.03 |
| DEPF | 0.00 ±0.00 | -2.17e6.00 ±1.26e4.00 | -2.17e6.00 ±1.26e4.00 | -2.17e6.00 ±1.26e4.00 | -2.17e6.00 ±1.26e4.00 | 0.00 ±0.00 | -2.19e6.00 ±1.76e4.00 | -2.19e6.00 ±1.76e4.00 | -2.19e6.00 ±1.76e4.00 | -2.19e6.00 ±1.76e4.00 | 0.00 ±0.00 | -2.19e6.00 ±1.31e4.00 | -2.19e6.00 ±1.31e4.00 | -2.19e6.00 ±1.31e4.00 | -2.19e6.00 ±1.31e4.00 |
| DESPF | 0.00 ±0.00 | -2.17e6.00 ±1.26e4.00 | -2.17e6.00 ±1.26e4.00 | -2.17e6.00 ±1.26e4.00 | -2.17e6.00 ±1.26e4.00 | 0.00 ±0.00 | -2.19e6.00 ±1.76e4.00 | -2.19e6.00 ±1.76e4.00 | -2.19e6.00 ±1.76e4.00 | -2.19e6.00 ±1.76e4.00 | 0.00 ±0.00 | -2.19e6.00 ±1.31e4.00 | -2.19e6.00 ±1.31e4.00 | -2.19e6.00 ±1.31e4.00 | -2.19e6.00 ±1.31e4.00 |
| PRIME | 0.00 ±0.00 | 0.16 ±0.23 | 0.32 ±0.46 | 0.21 ±0.31 | 0.24 ±0.35 | 0.00 ±0.00 | 0.14 ±0.24 | 0.28 ±0.47 | 0.18 ±0.31 | 0.17 ±0.28 | 0.00 ±0.00 | 0.13 ±0.20 | 0.26 ±0.47 | 0.17 ±0.31 | 0.13 ±0.20 |
| Task | | | | | | | | GTOPX 7 | | | | | | | |
| $f(x^*_{DEF})$ | | | | | | | | -424.06 ±5.00 | | | | | | | |
| CARCOO | -384.82 ±107.07 | 0.82 ±0.09 | 1.14 ±0.30 | 0.93 ±0.12 | 0.98 ±0.13 | -423.80 ±3.98 | 0.79 ±0.22 | 1.08 ±0.12 | 0.88 ±0.13 | 0.85 ±0.16 | -423.80 ±4.99 | 0.77 ±0.23 | 1.05 ±0.08 | 0.87 ±0.16 | 0.78 ±0.23 |
| DEPF | -1.05e3.00 ±7.94 | -4.54e4.00 ±2.61e4.00 | -4.54e4.00 ±2.61e4.00 | -4.54e4.00 ±2.61e4.00 | -4.54e4.00 ±2.61e4.00 | -935.27 ±199.10 | -4.44e4.00 ±2.08e4.00 | -4.44e4.00 ±2.08e4.00 | -4.44e4.00 ±2.08e4.00 | -4.44e4.00 ±2.08e4.00 | -1.05e3.00 ±7.94 | -4.36e4.00 ±2.62e4.00 | -4.36e4.00 ±2.62e4.00 | -4.36e4.00 ±2.62e4.00 | -4.36e4.00 ±2.62e4.00 |
| DESPF | -1.05e3.00 ±7.94 | -6.01e4.00 ±2.13e4.00 | -6.01e4.00 ±2.13e4.00 | -6.01e4.00 ±2.13e4.00 | -6.01e4.00 ±2.13e4.00 | -1.05e3.00 ±7.94 | -6.12e4.00 ±1.09e4.00 | -6.12e4.00 ±1.09e4.00 | -6.12e4.00 ±1.09e4.00 | -6.12e4.00 ±1.09e4.00 | -1.05e3.00 ±7.73 | -6.16e4.00 ±1.79e4.00 | -6.16e4.00 ±1.79e4.00 | -6.16e4.00 ±1.79e4.00 | -6.16e4.00 ±1.79e4.00 |
| PRIME | -1.05e3.00 ±7.94 | 0.02 ±0.11 | 0.04 ±0.23 | 0.03 ±0.15 | 0.03 ±0.17 | -1.05e3.00 ±7.94 | -0.17 ±0.08 | -0.34 ±0.17 | -0.23 ±0.11 | -0.21 ±0.10 | -1.05e3.00 ±7.94 | -0.24 ±0.08 | -0.47 ±0.15 | -0.32 ±0.10 | -0.24 ±0.08 |

Table 101: Overall results for GTOPX constrained tasks with 128 solutions and 100th percentile evaluations. In this case, 30% of the values are missing near the worst value and another 20% near the optimal value. Details are the same as Table 5.

| Steps | | t = 50 | | | | | t = 100 | | | | | t = 150 | | | |
|---|---|---|---|---|---|---|---|---|---|---|---|---|---|---|---|
| Task | | | | | | | GTOPX 1 | | | | | | | | |
| $f(x^*_{DIFF})$ | | | | | | | -64.89$_{\pm 0.08}$ | | | | | | | | |
| Metric | FS | SI | OI | SO | SO$_\omega$ | FS | SI | OI | SO | SO$_\omega$ | FS | SI | OI | SO | SO$_\omega$ |
| CARCOO | -63.88$_{\pm 2.16}$ | 0.88$_{\pm 0.15}$ | 1.11$_{\pm 0.13}$ | 0.97$_{\pm 0.11}$ | 1.01$_{\pm 0.10}$ | -63.06$_{\pm 3.71}$ | 0.81$_{\pm 0.17}$ | 1.20$_{\pm 0.15}$ | 0.95$_{\pm 0.12}$ | 0.89$_{\pm 0.14}$ | -62.47$_{\pm 4.07}$ | 0.79$_{\pm 0.17}$ | 1.25$_{\pm 0.18}$ | 0.95$_{\pm 0.12}$ | 0.79$_{\pm 0.17}$ |
| DEPF | -196.63$_{\pm 67.55}$ | -1.67e3.00$_{\pm 4.15e5.00}$ | -1.67e3.00$_{\pm 4.15e5.00}$ | -1.67e3.00$_{\pm 4.15e5.00}$ | -1.67e3.00$_{\pm 4.15e5.00}$ | -222.73$_{\pm 32.53}$ | -1.89e3.00$_{\pm 5.00e5.00}$ | -1.90e3.00$_{\pm 5.00e5.00}$ | -1.90e3.00$_{\pm 5.00e5.00}$ | -1.89e3.00$_{\pm 5.00e5.00}$ | -235.24$_{\pm 2.93}$ | -1.97e3.00$_{\pm 5.20e5.00}$ | -1.97e3.00$_{\pm 5.20e5.00}$ | -1.97e3.00$_{\pm 5.20e5.00}$ | -1.97e3.00$_{\pm 5.20e5.00}$ |
| DESPF | -235.24$_{\pm 2.93}$ | -1.26e4.00$_{\pm 7.26e5.00}$ | -1.26e4.00$_{\pm 7.26e5.00}$ | -1.26e4.00$_{\pm 7.26e5.00}$ | -1.26e4.00$_{\pm 7.26e5.00}$ | -235.24$_{\pm 2.93}$ | -1.27e4.00$_{\pm 7.32e5.00}$ | -1.27e4.00$_{\pm 7.32e5.00}$ | -1.27e4.00$_{\pm 7.32e5.00}$ | -1.27e4.00$_{\pm 7.32e5.00}$ | -235.24$_{\pm 2.93}$ | -1.27e4.00$_{\pm 7.36e5.00}$ | -1.27e4.00$_{\pm 7.36e5.00}$ | -1.27e4.00$_{\pm 7.36e5.00}$ | -1.27e4.00$_{\pm 7.36e5.00}$ |
| PRIME | -235.24$_{\pm 2.93}$ | -1.07$_{\pm 0.60}$ | -2.14$_{\pm 1.39}$ | -1.43$_{\pm 0.80}$ | -1.61$_{\pm 0.90}$ | -235.24$_{\pm 2.93}$ | -1.97$_{\pm 0.76}$ | -3.95$_{\pm 3.11}$ | -2.63$_{\pm 0.74}$ | -2.37$_{\pm 0.67}$ | -235.24$_{\pm 2.93}$ | -2.27$_{\pm 0.57}$ | -4.54$_{\pm 1.15}$ | -3.03$_{\pm 0.76}$ | -2.28$_{\pm 0.57}$ |
| Task | | | | | | | GTOPX 5 | | | | | | | | |
| $f(x^*_{DIFF})$ | | | | | | | 5.39e4$_{\pm 2.0e3.00}$ | | | | | | | | |
| CARCOO | 1.18e5.00$_{\pm 3.63e4.00}$ | 0.78$_{\pm 0.05}$ | 1.46$_{\pm 0.10}$ | 1.02$_{\pm 0.05}$ | 1.13$_{\pm 0.06}$ | 9.94e4.00$_{\pm 3.44e4.00}$ | 0.75$_{\pm 0.06}$ | 1.45$_{\pm 0.11}$ | 0.99$_{\pm 0.07}$ | 0.90$_{\pm 0.07}$ | 7.53e4.00$_{\pm 3.94e4.00}$ | 0.70$_{\pm 0.07}$ | 1.41$_{\pm 0.14}$ | 0.94$_{\pm 0.09}$ | 0.71$_{\pm 0.07}$ |
| DEPF | 0.00$_{\pm 0.00}$ | -5.34e6.00$_{\pm 1.98e5.00}$ | -5.34e6.00$_{\pm 1.98e5.00}$ | -5.34e6.00$_{\pm 1.98e5.00}$ | -5.34e6.00$_{\pm 1.98e5.00}$ | 0.00$_{\pm 0.00}$ | -5.37e6.00$_{\pm 1.99e5.00}$ | -5.37e6.00$_{\pm 1.99e5.00}$ | -5.37e6.00$_{\pm 1.99e5.00}$ | -5.37e6.00$_{\pm 1.99e5.00}$ | 0.00$_{\pm 0.00}$ | -5.38e6.00$_{\pm 1.99e5.00}$ | -5.38e6.00$_{\pm 1.99e5.00}$ | -5.38e6.00$_{\pm 1.99e5.00}$ | -5.38e6.00$_{\pm 1.99e5.00}$ |
| DESPF | 0.00$_{\pm 0.00}$ | -5.34e6.00$_{\pm 1.98e5.00}$ | -5.34e6.00$_{\pm 1.98e5.00}$ | -5.34e6.00$_{\pm 1.98e5.00}$ | -5.34e6.00$_{\pm 1.98e5.00}$ | 0.00$_{\pm 0.00}$ | -5.37e6.00$_{\pm 1.99e5.00}$ | -5.37e6.00$_{\pm 1.99e5.00}$ | -5.37e6.00$_{\pm 1.99e5.00}$ | -5.37e6.00$_{\pm 1.99e5.00}$ | 0.00$_{\pm 0.00}$ | -5.38e6.00$_{\pm 1.99e5.00}$ | -5.38e6.00$_{\pm 1.99e5.00}$ | -5.38e6.00$_{\pm 1.99e5.00}$ | -5.38e6.00$_{\pm 1.99e5.00}$ |
| PRIME | 0.00$_{\pm 0.00}$ | 0.10$_{\pm 0.23}$ | 0.20$_{\pm 0.47}$ | 0.13$_{\pm 0.31}$ | 0.15$_{\pm 0.35}$ | 0.00$_{\pm 0.00}$ | 0.08$_{\pm 0.24}$ | 0.15$_{\pm 0.48}$ | 0.10$_{\pm 0.32}$ | 0.09$_{\pm 0.29}$ | 0.00$_{\pm 0.00}$ | 0.07$_{\pm 0.24}$ | 0.14$_{\pm 0.49}$ | 0.09$_{\pm 0.31}$ | 0.07$_{\pm 0.25}$ |
| Task | | | | | | | GTOPX 7 | | | | | | | | |
| $f(x^*_{DIFF})$ | | | | | | | -276.88$_{\pm 4.27}$ | | | | | | | | |
| CARCOO | -360.70$_{\pm 200.73}$ | 0.69$_{\pm 0.26}$ | 1.00$_{\pm 0.26}$ | 0.79$_{\pm 0.24}$ | 0.84$_{\pm 0.24}$ | -360.70$_{\pm 200.73}$ | 0.60$_{\pm 0.19}$ | 0.83$_{\pm 0.32}$ | 0.68$_{\pm 0.41}$ | 0.65$_{\pm 0.40}$ | -360.70$_{\pm 200.73}$ | 0.57$_{\pm 0.43}$ | 0.78$_{\pm 0.46}$ | 0.64$_{\pm 0.47}$ | 0.58$_{\pm 0.43}$ |
| DEPF | -891.06$_{\pm 3.44}$ | -2.20e4.00$_{\pm 2.64e4.00}$ | -2.20e4.00$_{\pm 2.64e4.00}$ | -2.20e4.00$_{\pm 2.64e4.00}$ | -2.20e4.00$_{\pm 2.64e4.00}$ | -789.99$_{\pm 245.86}$ | -2.25e4.00$_{\pm 2.91e4.00}$ | -2.25e4.00$_{\pm 2.91e4.00}$ | -2.25e4.00$_{\pm 2.91e4.00}$ | -2.25e4.00$_{\pm 2.91e4.00}$ | -891.06$_{\pm 3.44}$ | -2.27e4.00$_{\pm 2.93e4.00}$ | -2.27e4.00$_{\pm 2.93e4.00}$ | -2.27e4.00$_{\pm 2.93e4.00}$ | -2.27e4.00$_{\pm 2.93e4.00}$ |
| DESPF | -891.06$_{\pm 3.44}$ | -4.54e4.00$_{\pm 2.62e4.00}$ | -4.54e4.00$_{\pm 2.62e4.00}$ | -4.54e4.00$_{\pm 2.62e4.00}$ | -4.54e4.00$_{\pm 2.62e4.00}$ | -891.06$_{\pm 3.44}$ | -4.57e4.00$_{\pm 2.64e4.00}$ | -4.57e4.00$_{\pm 2.64e4.00}$ | -4.57e4.00$_{\pm 2.64e4.00}$ | -4.57e4.00$_{\pm 2.64e4.00}$ | -891.06$_{\pm 3.44}$ | -4.58e4.00$_{\pm 2.65e4.00}$ | -4.58e4.00$_{\pm 2.65e4.00}$ | -4.58e4.00$_{\pm 2.65e4.00}$ | -4.58e4.00$_{\pm 2.65e4.00}$ |
| PRIME | -891.06$_{\pm 3.44}$ | -1.28$_{\pm 1.21}$ | -2.55$_{\pm 2.42}$ | -1.70$_{\pm 1.61}$ | -1.92$_{\pm 1.82}$ | -891.06$_{\pm 3.44}$ | -1.62$_{\pm 1.26}$ | -3.23$_{\pm 2.53}$ | -2.15$_{\pm 1.69}$ | -1.94$_{\pm 1.52}$ | -891.06$_{\pm 3.44}$ | -1.73$_{\pm 1.28}$ | -3.45$_{\pm 2.57}$ | -2.30$_{\pm 1.71}$ | -1.73$_{\pm 1.29}$ |

Table 102: Overall results for GTOPX constrained tasks with 128 solutions and 100th percentile evaluations. In this case, 40% of the values are missing near the worst value and another 10% near the optimal value. Details are the same as Table 5.

| Steps | | t = 50 | | | | | t = 100 | | | | | t = 150 | | | |
|---|---|---|---|---|---|---|---|---|---|---|---|---|---|---|---|
| Task | | | | | | | GTOPX 1 | | | | | | | | |
| $f(x^*_{DIFF})$ | | | | | | | -47.69$_{\pm 0.03}$ | | | | | | | | |
| Metric | FS | SI | OI | SO | SO$_\omega$ | FS | SI | OI | SO | SO$_\omega$ | FS | SI | OI | SO | SO$_\omega$ |
| CARCOO | -46.81$_{\pm 2.44}$ | 0.66$_{\pm 0.40}$ | 0.78$_{\pm 0.63}$ | 0.70$_{\pm 0.40}$ | 0.72$_{\pm 0.64}$ | -46.02$_{\pm 2.59}$ | 0.63$_{\pm 0.44}$ | 0.90$_{\pm 0.78}$ | 0.73$_{\pm 0.62}$ | 0.69$_{\pm 0.64}$ | -45.69$_{\pm 2.49}$ | 0.66$_{\pm 0.34}$ | 1.07$_{\pm 0.56}$ | 0.79$_{\pm 0.39}$ | 0.66$_{\pm 0.34}$ |
| DEPF | -153.76$_{\pm 65.83}$ | -2.66e3.00$_{\pm 4.69e3.00}$ | -2.67e3.00$_{\pm 4.68e3.00}$ | -2.66e3.00$_{\pm 4.69e3.00}$ | -2.66e3.00$_{\pm 4.68e3.00}$ | -203.83$_{\pm 2.26}$ | -3.05e3.00$_{\pm 5.32e3.00}$ | -3.06e3.00$_{\pm 5.32e3.00}$ | -3.05e3.00$_{\pm 5.32e3.00}$ | -3.05e3.00$_{\pm 5.32e3.00}$ | -203.83$_{\pm 2.24}$ | -3.33e3.00$_{\pm 5.77e3.00}$ | -3.34e3.00$_{\pm 5.77e3.00}$ | -3.33e3.00$_{\pm 5.77e3.00}$ | -3.33e3.00$_{\pm 5.77e3.00}$ |
| DESPF | -203.83$_{\pm 2.26}$ | -5.49e3.00$_{\pm 7.09e3.00}$ | -5.49e3.00$_{\pm 7.09e3.00}$ | -5.49e3.00$_{\pm 7.09e3.00}$ | -5.49e3.00$_{\pm 7.09e3.00}$ | -203.83$_{\pm 2.26}$ | -5.68e3.00$_{\pm 7.32e3.00}$ | -5.69e3.00$_{\pm 7.32e3.00}$ | -5.68e3.00$_{\pm 7.32e3.00}$ | -5.68e3.00$_{\pm 7.32e3.00}$ | -203.83$_{\pm 2.24}$ | -5.74e3.00$_{\pm 7.40e3.00}$ | -5.75e3.00$_{\pm 7.40e3.00}$ | -5.75e3.00$_{\pm 7.40e3.00}$ | -5.74e3.00$_{\pm 7.40e3.00}$ |
| PRIME | -203.83$_{\pm 2.26}$ | -1.54$_{\pm 0.51}$ | -3.09$_{\pm 1.02}$ | -2.06$_{\pm 0.68}$ | -2.32$_{\pm 0.77}$ | -203.83$_{\pm 2.26}$ | -2.39$_{\pm 0.51}$ | -4.79$_{\pm 1.01}$ | -3.19$_{\pm 0.67}$ | -2.88$_{\pm 0.61}$ | -203.83$_{\pm 2.24}$ | -2.67$_{\pm 0.52}$ | -5.35$_{\pm 1.04}$ | -3.56$_{\pm 0.69}$ | -2.68$_{\pm 0.52}$ |
| Task | | | | | | | GTOPX 5 | | | | | | | | |
| $f(x^*_{DIFF})$ | | | | | | | 1.10e5$_{\pm 3.54e3.00}$ | | | | | | | | |
| CARCOO | 1.20e5.00$_{\pm 1.00e4.00}$ | 0.66$_{\pm 0.01}$ | 1.32$_{\pm 0.09}$ | 0.88$_{\pm 0.06}$ | 0.99$_{\pm 0.07}$ | 1.10e5.00$_{\pm 3.54e4.00}$ | 0.61$_{\pm 0.06}$ | 1.21$_{\pm 0.12}$ | 0.81$_{\pm 0.08}$ | 0.73$_{\pm 0.07}$ | 1.11e5.00$_{\pm 3.66e3.00}$ | 0.58$_{\pm 0.05}$ | 1.16$_{\pm 0.11}$ | 0.77$_{\pm 0.07}$ | 0.58$_{\pm 0.05}$ |
| DEPF | 2.02$_{\pm 0.06}$ | -1.09e7.00$_{\pm 3.51e5.00}$ | -1.09e7.00$_{\pm 3.51e5.00}$ | -1.09e7.00$_{\pm 3.51e5.00}$ | -1.09e7.00$_{\pm 3.51e5.00}$ | 2.02$_{\pm 0.06}$ | -1.10e7.00$_{\pm 3.53e5.00}$ | -1.10e7.00$_{\pm 3.53e5.00}$ | -1.10e7.00$_{\pm 3.53e5.00}$ | -1.10e7.00$_{\pm 3.53e5.00}$ | 2.02$_{\pm 0.06}$ | -1.10e7.00$_{\pm 3.54e5.00}$ | -1.10e7.00$_{\pm 3.54e5.00}$ | -1.10e7.00$_{\pm 3.54e5.00}$ | -1.10e7.00$_{\pm 3.54e5.00}$ |
| DESPF | 2.02$_{\pm 0.06}$ | -1.09e7.00$_{\pm 3.51e5.00}$ | -1.09e7.00$_{\pm 3.51e5.00}$ | -1.09e7.00$_{\pm 3.51e5.00}$ | -1.09e7.00$_{\pm 3.51e5.00}$ | 2.02$_{\pm 0.06}$ | -1.10e7.00$_{\pm 3.53e5.00}$ | -1.10e7.00$_{\pm 3.53e5.00}$ | -1.10e7.00$_{\pm 3.53e5.00}$ | -1.10e7.00$_{\pm 3.53e5.00}$ | 2.02$_{\pm 0.06}$ | -1.10e7.00$_{\pm 3.54e5.00}$ | -1.10e7.00$_{\pm 3.54e5.00}$ | -1.10e7.00$_{\pm 3.54e5.00}$ | -1.10e7.00$_{\pm 3.54e5.00}$ |
| PRIME | 2.02$_{\pm 0.06}$ | -0.68$_{\pm 1.25}$ | -1.36$_{\pm 2.51}$ | -0.91$_{\pm 1.67}$ | -1.02$_{\pm 1.88}$ | 2.02$_{\pm 0.06}$ | -0.72$_{\pm 1.28}$ | -1.44$_{\pm 2.56}$ | -0.96$_{\pm 1.71}$ | -0.87$_{\pm 1.54}$ | 2.02$_{\pm 0.06}$ | -0.73$_{\pm 1.29}$ | -1.47$_{\pm 2.58}$ | -0.98$_{\pm 1.72}$ | -0.74$_{\pm 1.29}$ |
| Task | | | | | | | GTOPX 7 | | | | | | | | |
| $f(x^*_{DIFF})$ | | | | | | | -149.12$_{\pm 5.17}$ | | | | | | | | |
| CARCOO | -499.52$_{\pm 295.65}$ | 0.32$_{\pm 0.49}$ | 0.39$_{\pm 0.67}$ | 0.34$_{\pm 0.56}$ | 0.35$_{\pm 0.57}$ | -545.73$_{\pm 307.04}$ | -0.40$_{\pm 0.96}$ | -1.06$_{\pm 1.62}$ | -0.62$_{\pm 1.19}$ | -0.54$_{\pm 1.13}$ | -545.73$_{\pm 307.04}$ | -0.65$_{\pm 1.16}$ | -1.55$_{\pm 1.99}$ | -0.95$_{\pm 1.48}$ | -0.65$_{\pm 1.17}$ |
| DEPF | -781.52$_{\pm 4.29}$ | -3.59e4.00$_{\pm 2.02e4.00}$ | -3.59e4.00$_{\pm 2.02e4.00}$ | -3.59e4.00$_{\pm 2.02e4.00}$ | -3.59e4.00$_{\pm 2.02e4.00}$ | -774.48$_{\pm 19.02}$ | -3.67e4.00$_{\pm 2.06e4.00}$ | -3.67e4.00$_{\pm 2.06e4.00}$ | -3.67e4.00$_{\pm 2.06e4.00}$ | -3.67e4.00$_{\pm 2.06e4.00}$ | -781.52$_{\pm 4.29}$ | -3.77e4.00$_{\pm 2.92e4.00}$ | -3.77e4.00$_{\pm 2.92e4.00}$ | -3.77e4.00$_{\pm 2.92e4.00}$ | -3.77e4.00$_{\pm 2.92e4.00}$ |
| DESPF | -781.52$_{\pm 4.29}$ | -4.68e4.00$_{\pm 2.76e4.00}$ | -4.68e4.00$_{\pm 2.76e4.00}$ | -4.68e4.00$_{\pm 2.76e4.00}$ | -4.68e4.00$_{\pm 2.76e4.00}$ | -781.52$_{\pm 4.29}$ | -4.72e4.00$_{\pm 2.72e4.00}$ | -4.72e4.00$_{\pm 2.72e4.00}$ | -4.72e4.00$_{\pm 2.72e4.00}$ | -4.72e4.00$_{\pm 2.72e4.00}$ | -781.52$_{\pm 4.29}$ | -4.73e4.00$_{\pm 2.73e4.00}$ | -4.73e4.00$_{\pm 2.73e4.00}$ | -4.73e4.00$_{\pm 2.73e4.00}$ | -4.73e4.00$_{\pm 2.73e4.00}$ |
| PRIME | -781.52$_{\pm 4.29}$ | -1.99$_{\pm 0.41}$ | -3.98$_{\pm 0.83}$ | -2.65$_{\pm 0.55}$ | -2.99$_{\pm 0.62}$ | -781.52$_{\pm 4.29}$ | -2.51$_{\pm 0.49}$ | -5.02$_{\pm 0.81}$ | -3.35$_{\pm 0.74}$ | -3.02$_{\pm 0.49}$ | -781.52$_{\pm 4.29}$ | -2.68$_{\pm 0.51}$ | -5.36$_{\pm 0.62}$ | -3.57$_{\pm 0.54}$ | -2.69$_{\pm 0.41}$ |

Table 103: Overall results for GTOPX constrained tasks with 128 solutions and 50th percentile evaluations. In this case, 0% of the values are missing near the worst value and another 50% near the optimal value. Details are the same as Table 5.

| Steps | | t = 50 | | | | | t = 100 | | | | | t = 150 | | | |
|---|---|---|---|---|---|---|---|---|---|---|---|---|---|---|---|
| Task | | | | | | | GTOPX 1 | | | | | | | | |
| $f(x^*_{DIFF})$ | | | | | | | -164.33$_{\pm 3.48}$ | | | | | | | | |
| Metric | FS | SI | OI | SO | SO$_\omega$ | FS | SI | OI | SO | SO$_\omega$ | FS | SI | OI | SO | SO$_\omega$ |
| CARCOO | - | - | - | - | - | - | - | - | - | - | - | - | - | - | - |
| DEPF | -768.01$_{\pm 17.71}$ | -3.89$_{\pm 0.51}$ | -3.89$_{\pm 0.51}$ | -3.89$_{\pm 0.51}$ | -3.89$_{\pm 0.51}$ | -768.01$_{\pm 17.71}$ | -3.91$_{\pm 0.51}$ | -3.91$_{\pm 0.51}$ | -3.91$_{\pm 0.51}$ | -3.91$_{\pm 0.51}$ | -768.01$_{\pm 17.71}$ | -3.92$_{\pm 0.51}$ | -3.92$_{\pm 0.51}$ | -3.92$_{\pm 0.51}$ | -3.92$_{\pm 0.51}$ |
| DESPF | -768.01$_{\pm 17.71}$ | -7.00$_{\pm 3.70}$ | -7.00$_{\pm 3.70}$ | -7.00$_{\pm 3.70}$ | -7.00$_{\pm 3.70}$ | -768.01$_{\pm 17.71}$ | -7.04$_{\pm 3.72}$ | -7.04$_{\pm 3.72}$ | -7.04$_{\pm 3.72}$ | -7.04$_{\pm 3.72}$ | -768.01$_{\pm 17.71}$ | -7.06$_{\pm 3.72}$ | -7.06$_{\pm 3.72}$ | -7.06$_{\pm 3.72}$ | -7.06$_{\pm 3.72}$ |
| PRIME | - | - | - | - | - | - | - | - | - | - | - | - | - | - | - |
| Task | | | | | | | GTOPX 5 | | | | | | | | |
| $f(x^*_{DIFF})$ | | | | | | | 6.41$_{\pm 1.48}$ | | | | | | | | |
| CARCOO | - | - | - | - | - | - | - | - | - | - | - | - | - | - | - |
| DEPF | 0.00$_{\pm 0.00}$ | -632.95$_{\pm 187.40}$ | -632.95$_{\pm 187.40}$ | -632.95$_{\pm 187.40}$ | -632.95$_{\pm 187.40}$ | 0.00$_{\pm 0.00}$ | -636.25$_{\pm 188.58}$ | -636.25$_{\pm 188.58}$ | -636.25$_{\pm 188.58}$ | -636.25$_{\pm 188.58}$ | 0.00$_{\pm 0.00}$ | -637.33$_{\pm 188.70}$ | -637.33$_{\pm 188.70}$ | -637.33$_{\pm 188.70}$ | -637.33$_{\pm 188.70}$ |
| DESPF | 0.00$_{\pm 0.00}$ | -632.95$_{\pm 187.40}$ | -632.95$_{\pm 187.40}$ | -632.95$_{\pm 187.40}$ | -632.95$_{\pm 187.40}$ | 0.00$_{\pm 0.00}$ | -636.25$_{\pm 188.58}$ | -636.25$_{\pm 188.58}$ | -636.25$_{\pm 188.58}$ | -636.25$_{\pm 188.58}$ | 0.00$_{\pm 0.00}$ | -637.33$_{\pm 188.70}$ | -637.33$_{\pm 188.70}$ | -637.33$_{\pm 188.70}$ | -637.33$_{\pm 188.70}$ |
| PRIME | - | - | - | - | - | - | - | - | - | - | - | - | - | - | - |
| Task | | | | | | | GTOPX 7 | | | | | | | | |
| $f(x^*_{DIFF})$ | | | | | | | -670.34$_{\pm 3.13}$ | | | | | | | | |
| CARCOO | -2.04e3.00$_{\pm 1.75e4.00}$ | -240.11$_{\pm 414.64}$ | -240.11$_{\pm 410.44}$ | -240.11$_{\pm 410.44}$ | -240.11$_{\pm 410.44}$ | -2.04e3.00$_{\pm 1.75e4.00}$ | -241.44$_{\pm 416.55}$ | -241.44$_{\pm 416.55}$ | -241.44$_{\pm 416.55}$ | -241.44$_{\pm 416.55}$ | -2.04e3.00$_{\pm 1.75e4.00}$ | -241.88$_{\pm 417.25}$ | -241.88$_{\pm 417.25}$ | -241.88$_{\pm 417.25}$ | -241.88$_{\pm 417.25}$ |
| DEPF | -4.37e3.00$_{\pm 881.68}$ | -1.70e5.00$_{\pm 1.76e5.00}$ | -1.70e5.00$_{\pm 1.76e5.00}$ | -1.70e5.00$_{\pm 1.76e5.00}$ | -1.70e5.00$_{\pm 1.76e5.00}$ | -4.37e3.00$_{\pm 881.68}$ | -1.71e5.00$_{\pm 1.76e5.00}$ | -1.71e5.00$_{\pm 1.76e5.00}$ | -1.71e5.00$_{\pm 1.76e5.00}$ | -1.71e5.00$_{\pm 1.76e5.00}$ | -4.37e3.00$_{\pm 881.68}$ | -1.72e5.00$_{\pm 1.76e5.00}$ | -1.72e5.00$_{\pm 1.76e5.00}$ | -1.72e5.00$_{\pm 1.76e5.00}$ | -1.72e5.00$_{\pm 1.76e5.00}$ |
| DESPF | -4.32e3.00$_{\pm 496.57}$ | -1.88e5.00$_{\pm 1.98e5.00}$ | -1.88e5.00$_{\pm 1.98e5.00}$ | -1.88e5.00$_{\pm 1.98e5.00}$ | -1.88e5.00$_{\pm 1.98e5.00}$ | -4.32e3.00$_{\pm 496.57}$ | -1.89e5.00$_{\pm 1.95e5.00}$ | -1.89e5.00$_{\pm 1.95e5.00}$ | -1.89e5.00$_{\pm 1.95e5.00}$ | -1.89e5.00$_{\pm 1.95e5.00}$ | -4.32e3.00$_{\pm 496.57}$ | -1.89e5.00$_{\pm 1.92e5.00}$ | -1.89e5.00$_{\pm 1.92e5.00}$ | -1.89e5.00$_{\pm 1.92e5.00}$ | -1.89e5.00$_{\pm 1.92e5.00}$ |
| PRIME | -4.32e3.00$_{\pm 496.57}$ | -10.48$_{\pm 12.16}$ | -10.48$_{\pm 12.16}$ | -10.48$_{\pm 12.16}$ | -10.48$_{\pm 12.16}$ | -4.32e3.00$_{\pm 496.57}$ | -10.59$_{\pm 12.30}$ | -10.59$_{\pm 12.30}$ | -10.59$_{\pm 12.30}$ | -10.59$_{\pm 12.30}$ | -4.32e3.00$_{\pm 496.57}$ | -10.62$_{\pm 12.21}$ | -10.62$_{\pm 12.21}$ | -10.62$_{\pm 12.21}$ | -10.62$_{\pm 12.21}$ |

Table 104: Overall results for GTOPX constrained tasks with 128 solutions and 50th percentile evaluations. In this case, 0% of the values are missing near the worst value and another 40% near the optimal value. Details are the same as Table 5.

Table 105: Overall results for GTOPX constrained tasks with 128 solutions and 50th percentile evaluations. In this case, 0% of the values are missing near the worst value and another 30% near the optimal value. Details are the same as Table 5.

Table 106: Overall results for GTOPX constrained tasks with 128 solutions and 50th percentile evaluations. In this case, 0% of the values are missing near the worst value and another 20% near the optimal value. Details are the same as Table 5.

Table 107: Overall results for GTOPX constrained tasks with 128 solutions and 50th percentile evaluations. In this case, 0% of the values are missing near the worst value and another 10% near the optimal value. Details are the same as Table 5.

Table 108: Overall results for GTOPX constrained tasks with 128 solutions and 50th percentile evaluations. In this case, 10% of the values are missing near the worst value and another 40% near the optimal value. Details are the same as Table 5.

Table 109: Overall results for GTOPX constrained tasks with 128 solutions and 50th percentile evaluations. In this case, 20% of the values are missing near the worst value and another 30% near the optimal value. Details are the same as Table 5.

Table 110: Overall results for GTOPX constrained tasks with 128 solutions and 50th percentile evaluations. In this case, 25% of the values are missing near the worst value and another 25% near the optimal value. Details are the same as Table 5.

Table 111: Overall results for GTOPX constrained tasks with 128 solutions and 50th percentile evaluations. In this case, 30% of the values are missing near the worst value and another 20% near the optimal value. Details are the same as Table 5.

Table 112: Overall results for GTOPX constrained tasks with 128 solutions and 50th percentile evaluations. In this case, 40% of the values are missing near the worst value and another 10% near the optimal value. Details are the same as Table 5.

Table 113: Overall results for GTOPX constrained tasks with 128 solutions and 0th percentile evaluations. In this case, 0% of the values are missing near the worst value and another 50% near the optimal value. Details are the same as Table 5.

Table 114: Overall results for GTOPX constrained tasks with 128 solutions and 0th percentile evaluations. In this case, 0% of the values are missing near the worst value and another 40% near the optimal value. Details are the same as Table 5.

Table 115: Overall results for GTOPX constrained tasks with 128 solutions and 0th percentile evaluations. In this case, 0% of the values are missing near the worst value and another 30% near the optimal value. Details are the same as Table 5.

Table 116: Overall results for GTOPX constrained tasks with 128 solutions and 0th percentile evaluations. In this case, 0% of the values are missing near the worst value and another 20% near the optimal value. Details are the same as Table 5.

Table 117: Overall results for GTOPX constrained tasks with 128 solutions and 0th percentile evaluations. In this case, 0% of the values are missing near the worst value and another 10% near the optimal value. Details are the same as Table 5.

Table 118: Overall results for GTOPX constrained tasks with 128 solutions and 0th percentile evaluations. In this case, 10% of the values are missing near the worst value and another 40% near the optimal value. Details are the same as Table 5.

Table 119: Overall results for GTOPX constrained tasks with 128 solutions and 0th percentile evaluations. In this case, 20% of the values are missing near the worst value and another 30% near the optimal value. Details are the same as Table 5.

| Steps | t = 50 | | | | | t = 100 | | | | | t = 150 | | | | |
|---|---|---|---|---|---|---|---|---|---|---|---|---|---|---|---|
| Task | GTOPX 1 | | | | | | | | | | | | | | |
| $f(x^*_{DEPF})$ | -86.91±0.50 | | | | | | | | | | | | | | |
| Metric | FS | SI | OI | SO | SO_x | FS | SI | OI | SO | SO_x | FS | SI | OI | SO | SO_x |
| CARCOO | -279.91±145.33 | -290.61±282.83 | -290.61±282.83 | -290.61±282.83 | -290.61±282.83 | -279.91±145.33 | -305.85±278.42 | -305.85±278.42 | -305.85±278.42 | -305.85±278.42 | -279.91±145.33 | -310.86±277.32 | -310.86±277.32 | -310.86±277.32 | -310.86±277.32 |
| DEPF | -389.73±11.94 | -18.02±29.65 | -18.02±29.65 | -18.02±29.65 | -18.02±29.65 | -389.73±11.94 | -18.12±29.81 | -18.12±29.81 | -18.12±29.81 | -18.12±29.81 | -389.73±11.94 | -18.15±29.86 | -18.15±29.86 | -18.15±29.86 | -18.15±29.86 |
| DESPF | -389.73±11.96 | -1.09e4.00±1.41e4.00 | -1.09e4.00±1.41e4.00 | -1.09e4.00±1.41e4.00 | -1.09e4.00±1.41e4.00 | -389.73±11.96 | -1.10e4.00±1.43e4.00 | -1.10e4.00±1.43e4.00 | -1.10e4.00±1.43e4.00 | -1.10e4.00±1.43e4.00 | -389.73±11.96 | -1.10e4.00±1.43e4.00 | -1.10e4.00±1.43e4.00 | -1.10e4.00±1.43e4.00 | -1.10e4.00±1.43e4.00 |
| PRIME | -389.73±11.96 | -4.57±0.77 | -4.57±0.77 | -4.57±0.77 | -4.57±0.77 | -389.73±11.96 | -4.60±0.77 | -4.60±0.77 | -4.60±0.77 | -4.60±0.77 | -389.73±11.96 | -4.61±0.77 | -4.61±0.77 | -4.61±0.77 | -4.61±0.77 |
| Task | GTOPX 5 | | | | | | | | | | | | | | |
| $f(x^*_{DEPF})$ | 2.20e4±751.38 | | | | | | | | | | | | | | |
| CARCOO | 0.00±0.00 | 0.49±0.19 | 0.49±0.19 | 0.49±0.19 | 0.49±0.19 | 0.00±0.00 | 0.47±0.21 | 0.47±0.21 | 0.47±0.21 | 0.47±0.21 | 0.00±0.00 | 0.47±0.22 | 0.47±0.22 | 0.47±0.22 | 0.47±0.22 |
| DEPF | 0.00±0.00 | -2.17e6.00±7.26e4.00 | -2.17e6.00±7.26e4.00 | -2.17e6.00±7.26e4.00 | -2.17e6.00±7.26e4.00 | 0.00±0.00 | -2.19e6.00±7.39e4.00 | -2.19e6.00±7.39e4.00 | -2.19e6.00±7.39e4.00 | -2.19e6.00±7.39e4.00 | 0.00±0.00 | -2.19e6.00±7.31e4.00 | -2.19e6.00±7.31e4.00 | -2.19e6.00±7.31e4.00 | -2.19e6.00±7.31e4.00 |
| DESPF | 0.00±0.00 | -2.17e6.00±7.26e4.00 | -2.17e6.00±7.26e4.00 | -2.17e6.00±7.26e4.00 | -2.17e6.00±7.26e4.00 | 0.00±0.00 | -2.19e6.00±7.39e4.00 | -2.19e6.00±7.39e4.00 | -2.19e6.00±7.39e4.00 | -2.19e6.00±7.39e4.00 | 0.00±0.00 | -2.19e6.00±7.31e4.00 | -2.19e6.00±7.31e4.00 | -2.19e6.00±7.31e4.00 | -2.19e6.00±7.31e4.00 |
| PRIME | 0.00±0.00 | 0.24±0.48 | 0.24±0.48 | 0.24±0.48 | 0.24±0.48 | 0.00±0.00 | 0.23±0.48 | 0.23±0.48 | 0.23±0.48 | 0.23±0.48 | 0.00±0.00 | 0.23±0.48 | 0.23±0.48 | 0.23±0.48 | 0.23±0.48 |
| Task | GTOPX 7 | | | | | | | | | | | | | | |
| $f(x^*_{DEPF})$ | -424.06±5.09 | | | | | | | | | | | | | | |
| CARCOO | -826.96±282.05 | -9.00e3.00±1.97e4.00 | -9.00e3.00±1.97e4.00 | -9.00e3.00±1.97e4.00 | -9.00e3.00±1.97e4.00 | -826.96±282.05 | -9.05e3.00±1.98e4.00 | -9.05e3.00±1.98e4.00 | -9.05e3.00±1.98e4.00 | -9.05e3.00±1.98e4.00 | -826.96±282.05 | -9.07e3.00±1.98e4.00 | -9.07e3.00±1.98e4.00 | -9.07e3.00±1.98e4.00 | -9.07e3.00±1.98e4.00 |
| DEPF | -1.05e3.00±7.73 | -6.18e4.00±7.66.06 | -6.18e4.00±7.66.06 | -6.18e4.00±7.66.06 | -6.18e4.00±7.66.06 | -1.05e3.00±7.73 | -6.21e4.00±6.45.62 | -6.21e4.00±6.45.62 | -6.21e4.00±6.45.62 | -6.21e4.00±6.45.62 | -1.05e3.00±7.73 | -6.21e4.00±6.05.56 | -6.21e4.00±6.05.56 | -6.21e4.00±6.05.56 | -6.21e4.00±6.05.56 |
| PRIME | -1.05e3.00±7.86 | -0.74±0.15 | -0.74±0.15 | -0.74±0.15 | -0.74±0.15 | -1.05e3.00±7.86 | -0.73±0.14 | -0.73±0.14 | -0.73±0.14 | -0.73±0.14 | -1.05e3.00±7.86 | -0.73±0.14 | -0.73±0.14 | -0.73±0.14 | -0.73±0.14 |

Table 120: Overall results for GTOPX constrained tasks with 128 solutions and 0th percentile evaluations. In this case, 25% of the values are missing near the worst value and another 25% near the optimal value. Details are the same as Table 5.

| Steps | t = 50 | | | | | t = 100 | | | | | t = 150 | | | | |
|---|---|---|---|---|---|---|---|---|---|---|---|---|---|---|---|
| Task | GTOPX 1 | | | | | | | | | | | | | | |
| $f(x^*_{DEPF})$ | -75.14±0.98 | | | | | | | | | | | | | | |
| Metric | FS | SI | OI | SO | SO_x | FS | SI | OI | SO | SO_x | FS | SI | OI | SO | SO_x |
| CARCOO | -133.95±62.86 | -488.75±259.08 | -488.75±259.08 | -488.75±259.08 | -488.75±259.08 | -159.04±101.52 | -1.98e3.00±3.38e3.00 | -1.98e3.00±3.38e3.00 | -1.98e3.00±3.38e3.00 | -1.98e3.00±3.38e3.00 | -209.23±160.67 | -3.20e3.00±4.57e3.00 | -3.20e3.00±4.57e3.00 | -3.20e3.00±4.57e3.00 | -3.20e3.00±4.57e3.00 |
| DEPF | -283.39±14.47 | -2.88e3.00±7.52e3.00 | -2.88e3.00±7.52e3.00 | -2.88e3.00±7.52e3.00 | -2.88e3.00±7.52e3.00 | -283.39±14.47 | -2.90e3.00±7.56e3.00 | -2.90e3.00±7.56e3.00 | -2.90e3.00±7.56e3.00 | -2.90e3.00±7.56e3.00 | -283.39±14.47 | -2.90e3.00±7.55e3.00 | -2.90e3.00±7.55e3.00 | -2.90e3.00±7.55e3.00 | -2.90e3.00±7.55e3.00 |
| DESPF | -283.39±14.47 | -1.02e4.00±1.03e4.00 | -1.02e4.00±1.03e4.00 | -1.02e4.00±1.03e4.00 | -1.02e4.00±1.03e4.00 | -283.39±14.47 | -1.02e4.00±1.03e4.00 | -1.02e4.00±1.03e4.00 | -1.02e4.00±1.03e4.00 | -1.02e4.00±1.03e4.00 | -283.39±14.47 | -1.02e4.00±1.03e4.00 | -1.02e4.00±1.03e4.00 | -1.02e4.00±1.03e4.00 | -1.02e4.00±1.03e4.00 |
| PRIME | -283.39±14.47 | -4.07±0.83 | -4.07±0.83 | -4.07±0.83 | -4.07±0.83 | -283.39±14.47 | -4.09±0.84 | -4.09±0.84 | -4.09±0.84 | -4.09±0.84 | -283.39±14.47 | -4.10±0.84 | -4.10±0.84 | -4.10±0.84 | -4.10±0.84 |
| Task | GTOPX 5 | | | | | | | | | | | | | | |
| $f(x^*_{DEPF})$ | 3.63e4±1.25e3.00 | | | | | | | | | | | | | | |
| CARCOO | 0.00±0.00 | 0.49±0.42 | 0.49±0.42 | 0.49±0.42 | 0.49±0.42 | 0.00±0.00 | 0.48±0.42 | 0.48±0.42 | 0.48±0.42 | 0.48±0.42 | 0.00±0.00 | 0.48±0.42 | 0.48±0.42 | 0.48±0.42 | 0.48±0.42 |
| DEPF | 0.00±0.00 | -3.59e6.00±1.24e5.00 | -3.59e6.00±1.24e5.00 | -3.59e6.00±1.24e5.00 | -3.59e6.00±1.24e5.00 | 0.00±0.00 | -3.61e6.00±1.24e5.00 | -3.61e6.00±1.24e5.00 | -3.61e6.00±1.24e5.00 | -3.61e6.00±1.24e5.00 | 0.00±0.00 | -3.61e6.00±1.24e5.00 | -3.61e6.00±1.24e5.00 | -3.61e6.00±1.24e5.00 | -3.61e6.00±1.24e5.00 |
| DESPF | 0.00±0.00 | -3.59e6.00±1.24e5.00 | -3.59e6.00±1.24e5.00 | -3.59e6.00±1.24e5.00 | -3.59e6.00±1.24e5.00 | 0.00±0.00 | -3.61e6.00±1.24e5.00 | -3.61e6.00±1.24e5.00 | -3.61e6.00±1.24e5.00 | -3.61e6.00±1.24e5.00 | 0.00±0.00 | -3.61e6.00±1.24e5.00 | -3.61e6.00±1.24e5.00 | -3.61e6.00±1.24e5.00 | -3.61e6.00±1.24e5.00 |
| PRIME | 0.00±0.00 | 0.14±0.71 | 0.14±0.71 | 0.14±0.71 | 0.14±0.71 | 0.00±0.00 | 0.13±0.72 | 0.13±0.72 | 0.13±0.72 | 0.13±0.72 | 0.00±0.00 | 0.13±0.72 | 0.13±0.72 | 0.13±0.72 | 0.13±0.72 |
| Task | GTOPX 7 | | | | | | | | | | | | | | |
| $f(x^*_{DEPF})$ | -346.52±6.31 | | | | | | | | | | | | | | |
| CARCOO | -884.16±186.39 | -8.16e3.00±1.99e4.00 | -8.16e3.00±1.99e4.00 | -8.16e3.00±1.99e4.00 | -8.16e3.00±1.99e4.00 | -884.16±186.39 | -8.20e3.00±2.08e4.00 | -8.20e3.00±2.08e4.00 | -8.20e3.00±2.08e4.00 | -8.20e3.00±2.08e4.00 | -884.16±186.39 | -8.22e3.00±2.01e4.00 | -8.22e3.00±2.01e4.00 | -8.22e3.00±2.01e4.00 | -8.22e3.00±2.01e4.00 |
| DEPF | -954.35±5.116 | -3.00e4.00±3.08e4.00 | -3.00e4.00±3.08e4.00 | -3.00e4.00±3.08e4.00 | -3.00e4.00±3.08e4.00 | -954.35±5.116 | -3.01e4.00±3.01e4.00 | -3.01e4.00±3.01e4.00 | -3.01e4.00±3.01e4.00 | -3.01e4.00±3.01e4.00 | -954.35±5.116 | -3.02e4.00±3.02e4.00 | -3.02e4.00±3.02e4.00 | -3.02e4.00±3.02e4.00 | -3.02e4.00±3.02e4.00 |
| DESPF | -954.35±5.116 | -4.51e4.00±2.62e4.00 | -4.51e4.00±2.62e4.00 | -4.51e4.00±2.62e4.00 | -4.51e4.00±2.62e4.00 | -954.35±5.116 | -4.54e4.00±2.62e4.00 | -4.54e4.00±2.62e4.00 | -4.54e4.00±2.62e4.00 | -4.54e4.00±2.62e4.00 | -954.35±5.116 | -4.55e4.00±2.62e4.00 | -4.55e4.00±2.62e4.00 | -4.55e4.00±2.62e4.00 | -4.55e4.00±2.62e4.00 |
| PRIME | -954.35±5.116 | -1.93±1.44 | -1.93±1.44 | -1.93±1.44 | -1.93±1.44 | -954.35±5.116 | -1.93±1.42 | -1.93±1.42 | -1.93±1.42 | -1.93±1.42 | -954.35±5.116 | -1.93±1.41 | -1.93±1.41 | -1.93±1.41 | -1.93±1.41 |

Table 121: Overall results for GTOPX constrained tasks with 128 solutions and 0th percentile evaluations. In this case, 30% of the values are missing near the worst value and another 20% near the optimal value. Details are the same as Table 5.

| Steps | t = 50 | | | | | t = 100 | | | | | t = 150 | | | | |
|---|---|---|---|---|---|---|---|---|---|---|---|---|---|---|---|
| Task | GTOPX 1 | | | | | | | | | | | | | | |
| $f(x^*_{DEPF})$ | -64.89±0.88 | | | | | | | | | | | | | | |
| Metric | FS | SI | OI | SO | SO_x | FS | SI | OI | SO | SO_x | FS | SI | OI | SO | SO_x |
| CARCOO | -173.40±80.79 | -168.99±174.15 | -168.99±174.15 | -168.99±174.15 | -168.99±174.15 | -173.40±80.79 | -173.59±172.59 | -173.59±172.59 | -173.59±172.59 | -173.59±172.59 | -173.40±80.79 | -175.11±172.09 | -175.11±172.09 | -175.11±172.09 | -175.11±172.09 |
| DEPF | -235.24±2.93 | -2.10e3.00±5.53e3.00 | -2.10e3.00±5.53e3.00 | -2.10e3.00±5.53e3.00 | -2.10e3.00±5.53e3.00 | -235.24±2.93 | -2.11e3.00±5.55e3.00 | -2.11e3.00±5.55e3.00 | -2.11e3.00±5.55e3.00 | -2.11e3.00±5.55e3.00 | -235.24±2.93 | -2.11e3.00±5.56e3.00 | -2.11e3.00±5.56e3.00 | -2.11e3.00±5.56e3.00 | -2.11e3.00±5.56e3.00 |
| DESPF | -235.24±2.93 | -1.26e4.00±7.29e3.00 | -1.26e4.00±7.29e3.00 | -1.26e4.00±7.29e3.00 | -1.26e4.00±7.29e3.00 | -235.24±2.93 | -1.27e4.00±7.33e3.00 | -1.27e4.00±7.33e3.00 | -1.27e4.00±7.33e3.00 | -1.27e4.00±7.33e3.00 | -235.24±2.93 | -1.27e4.00±7.34e3.00 | -1.27e4.00±7.34e3.00 | -1.27e4.00±7.34e3.00 | -1.27e4.00±7.34e3.00 |
| PRIME | -235.24±2.93 | -5.64±1.27 | -5.64±1.27 | -5.64±1.27 | -5.64±1.27 | -235.24±2.93 | -5.68±1.27 | -5.68±1.27 | -5.68±1.27 | -5.68±1.27 | -235.24±2.93 | -5.69±1.27 | -5.69±1.27 | -5.69±1.27 | -5.69±1.27 |
| Task | GTOPX 5 | | | | | | | | | | | | | | |
| $f(x^*_{DEPF})$ | 5.39e4±2.14e3.00 | | | | | | | | | | | | | | |
| CARCOO | 0.00±0.00 | 0.49±0.12 | 0.49±0.12 | 0.49±0.12 | 0.49±0.12 | 0.00±0.00 | 0.48±0.12 | 0.48±0.12 | 0.48±0.12 | 0.48±0.12 | 0.00±0.00 | 0.48±0.12 | 0.48±0.12 | 0.48±0.12 | 0.48±0.12 |
| DEPF | 0.00±0.00 | -5.34e6.00±1.98e5.00 | -5.34e6.00±1.98e5.00 | -5.34e6.00±1.98e5.00 | -5.34e6.00±1.98e5.00 | 0.00±0.00 | -5.37e6.00±1.99e5.00 | -5.37e6.00±1.99e5.00 | -5.37e6.00±1.99e5.00 | -5.37e6.00±1.99e5.00 | 0.00±0.00 | -5.38e6.00±1.99e5.00 | -5.38e6.00±1.99e5.00 | -5.38e6.00±1.99e5.00 | -5.38e6.00±1.99e5.00 |
| DESPF | 0.00±0.00 | -5.34e6.00±1.98e5.00 | -5.34e6.00±1.98e5.00 | -5.34e6.00±1.98e5.00 | -5.34e6.00±1.98e5.00 | 0.00±0.00 | -5.37e6.00±1.99e5.00 | -5.37e6.00±1.99e5.00 | -5.37e6.00±1.99e5.00 | -5.37e6.00±1.99e5.00 | 0.00±0.00 | -5.38e6.00±1.99e5.00 | -5.38e6.00±1.99e5.00 | -5.38e6.00±1.99e5.00 | -5.38e6.00±1.99e5.00 |
| PRIME | 0.00±0.00 | 0.12±0.50 | 0.12±0.50 | 0.12±0.50 | 0.12±0.50 | 0.00±0.00 | 0.11±0.50 | 0.11±0.50 | 0.11±0.50 | 0.11±0.50 | 0.00±0.00 | 0.11±0.50 | 0.11±0.50 | 0.11±0.50 | 0.11±0.50 |
| Task | GTOPX 7 | | | | | | | | | | | | | | |
| $f(x^*_{DEPF})$ | -276.88±4.27 | | | | | | | | | | | | | | |
| CARCOO | -750.83±243.77 | -8.98e3.00±1.98e4.00 | -8.98e3.00±1.98e4.00 | -8.98e3.00±1.98e4.00 | -8.98e3.00±1.98e4.00 | -750.83±243.77 | -9.02e3.00±1.99e4.00 | -9.02e3.00±1.99e4.00 | -9.02e3.00±1.99e4.00 | -9.02e3.00±1.99e4.00 | -750.83±243.77 | -9.04e3.00±1.99e4.00 | -9.04e3.00±1.99e4.00 | -9.04e3.00±1.99e4.00 | -9.04e3.00±1.99e4.00 |
| DEPF | -891.06±3.44 | -2.28e4.00±2.94e4.00 | -2.28e4.00±2.94e4.00 | -2.28e4.00±2.94e4.00 | -2.28e4.00±2.94e4.00 | -891.06±3.44 | -2.29e4.00±2.95e4.00 | -2.29e4.00±2.95e4.00 | -2.29e4.00±2.95e4.00 | -2.29e4.00±2.95e4.00 | -891.06±3.44 | -2.29e4.00±2.96e4.00 | -2.29e4.00±2.96e4.00 | -2.29e4.00±2.96e4.00 | -2.29e4.00±2.96e4.00 |
| DESPF | -891.06±3.44 | -4.59e4.00±2.65e4.00 | -4.59e4.00±2.65e4.00 | -4.59e4.00±2.65e4.00 | -4.59e4.00±2.65e4.00 | -891.06±3.44 | -4.60e4.00±2.66e4.00 | -4.60e4.00±2.66e4.00 | -4.60e4.00±2.66e4.00 | -4.60e4.00±2.66e4.00 | -891.06±3.44 | -4.60e4.00±2.66e4.00 | -4.60e4.00±2.66e4.00 | -4.60e4.00±2.66e4.00 | -4.60e4.00±2.66e4.00 |
| PRIME | -891.06±3.44 | -3.91±2.68 | -3.91±2.68 | -3.91±2.68 | -3.91±2.68 | -891.06±3.44 | -3.90±2.66 | -3.90±2.66 | -3.90±2.66 | -3.90±2.66 | -891.06±3.44 | -3.90±2.66 | -3.90±2.66 | -3.90±2.66 | -3.90±2.66 |

Table 122: Overall results for GTOPX constrained tasks with 128 solutions and 0th percentile evaluations. In this case, 40% of the values are missing near the worst value and another 10% near the optimal value. Details are the same as Table 5.

| Steps | t = 50 | | | | | t = 100 | | | | | t = 150 | | | | |
|---|---|---|---|---|---|---|---|---|---|---|---|---|---|---|---|
| Task | GTOPX 1 | | | | | | | | | | | | | | |
| $f(x^*_{DEPF})$ | -47.69±0.83 | | | | | | | | | | | | | | |
| Metric | FS | SI | OI | SO | SO_x | FS | SI | OI | SO | SO_x | FS | SI | OI | SO | SO_x |
| CARCOO | -127.89±76.16 | -1.08e3.00±1.94e3.00 | -1.08e3.00±1.94e3.00 | -1.08e3.00±1.94e3.00 | -1.08e3.00±1.94e3.00 | -147.25±73.70 | -1.78e3.00±3.53e3.00 | -1.78e3.00±3.53e3.00 | -1.78e3.00±3.53e3.00 | -1.78e3.00±3.53e3.00 | -185.19±93.67 | -2.02e3.00±4.05e3.00 | -2.02e3.00±4.05e3.00 | -2.02e3.00±4.05e3.00 | -2.02e3.00±4.05e3.00 |
| DEPF | -203.83±2.24 | -3.85e3.00±6.63e3.00 | -3.85e3.00±6.63e3.00 | -3.85e3.00±6.63e3.00 | -3.85e3.00±6.63e3.00 | -203.83±2.24 | -3.87e3.00±6.65e3.00 | -3.87e3.00±6.65e3.00 | -3.87e3.00±6.65e3.00 | -3.87e3.00±6.65e3.00 | -203.83±2.24 | -3.88e3.00±6.67e3.00 | -3.88e3.00±6.67e3.00 | -3.88e3.00±6.67e3.00 | -3.88e3.00±6.67e3.00 |
| DESPF | -203.83±2.24 | -5.82e3.00±7.48e3.00 | -5.82e3.00±7.48e3.00 | -5.82e3.00±7.48e3.00 | -5.82e3.00±7.48e3.00 | -203.83±2.24 | -5.85e3.00±7.52e3.00 | -5.85e3.00±7.52e3.00 | -5.85e3.00±7.52e3.00 | -5.85e3.00±7.52e3.00 | -203.83±2.24 | -5.86e3.00±7.53e3.00 | -5.86e3.00±7.53e3.00 | -5.86e3.00±7.53e3.00 | -5.86e3.00±7.53e3.00 |
| PRIME | -203.83±2.24 | -6.38±1.14 | -6.38±1.14 | -6.38±1.14 | -6.38±1.14 | -203.83±2.24 | -6.42±1.14 | -6.42±1.14 | -6.42±1.14 | -6.42±1.14 | -203.83±2.24 | -6.43±1.14 | -6.43±1.14 | -6.43±1.14 | -6.43±1.14 |
| Task | GTOPX 5 | | | | | | | | | | | | | | |
| $f(x^*_{DEPF})$ | 1.10e5±3.54e3.00 | | | | | | | | | | | | | | |
| CARCOO | 2.02±0.06 | 0.04±0.45 | 0.04±0.45 | 0.04±0.45 | 0.04±0.45 | 2.02±0.06 | 0.03±0.46 | 0.03±0.46 | 0.03±0.46 | 0.03±0.46 | 2.02±0.06 | 0.03±0.46 | 0.03±0.46 | 0.03±0.46 | 0.03±0.46 |
| DEPF | 2.02±0.06 | -1.09e7.00±3.51e5.00 | -1.09e7.00±3.51e5.00 | -1.09e7.00±3.51e5.00 | -1.09e7.00±3.51e5.00 | 2.02±0.06 | -1.10e7.00±3.53e5.00 | -1.10e7.00±3.53e5.00 | -1.10e7.00±3.53e5.00 | -1.10e7.00±3.53e5.00 | 2.02±0.06 | -1.10e7.00±3.55e5.00 | -1.10e7.00±3.55e5.00 | -1.10e7.00±3.55e5.00 | -1.10e7.00±3.55e5.00 |
| DESPF | 2.02±0.06 | -1.09e7.00±3.51e5.00 | -1.09e7.00±3.51e5.00 | -1.09e7.00±3.51e5.00 | -1.09e7.00±3.51e5.00 | 2.02±0.06 | -1.10e7.00±3.53e5.00 | -1.10e7.00±3.53e5.00 | -1.10e7.00±3.53e5.00 | -1.10e7.00±3.53e5.00 | 2.02±0.06 | -1.10e7.00±3.55e5.00 | -1.10e7.00±3.55e5.00 | -1.10e7.00±3.55e5.00 | -1.10e7.00±3.55e5.00 |
| PRIME | 2.02±0.06 | -1.50±2.38 | -1.50±2.38 | -1.50±2.38 | -1.50±2.38 | 2.02±0.06 | -1.51±2.40 | -1.51±2.40 | -1.51±2.40 | -1.51±2.40 | 2.02±0.06 | -1.51±2.40 | -1.51±2.40 | -1.51±2.40 | -1.51±2.40 |
| Task | GTOPX 7 | | | | | | | | | | | | | | |
| $f(x^*_{DEPF})$ | -149.12±3.37 | | | | | | | | | | | | | | |
| CARCOO | -641.02±246.56 | -1.64e3.00±3.03e3.00 | -1.64e3.00±3.03e3.00 | -1.64e3.00±3.03e3.00 | -1.64e3.00±3.03e3.00 | -641.02±246.56 | -1.65e3.00±2.85e3.00 | -1.65e3.00±2.85e3.00 | -1.65e3.00±2.85e3.00 | -1.65e3.00±2.85e3.00 | -641.02±246.56 | -1.65e3.00±2.85e3.00 | -1.65e3.00±2.85e3.00 | -1.65e3.00±2.85e3.00 | -1.65e3.00±2.85e3.00 |
| DEPF | -781.52±4.39 | -3.92e4.00±3.94e4.00 | -3.92e4.00±3.94e4.00 | -3.92e4.00±3.94e4.00 | -3.92e4.00±3.94e4.00 | -781.52±4.39 | -3.94e4.00±3.95e4.00 | -3.94e4.00±3.95e4.00 | -3.94e4.00±3.95e4.00 | -3.94e4.00±3.95e4.00 | -781.52±4.39 | -3.95e4.00±3.96e4.00 | -3.95e4.00±3.96e4.00 | -3.95e4.00±3.96e4.00 | -3.95e4.00±3.96e4.00 |
| DESPF | -781.52±4.39 | -5.18e4.00±3.24e4.00 | -5.18e4.00±3.24e4.00 | -5.18e4.00±3.24e4.00 | -5.18e4.00±3.24e4.00 | -1.13e3.00±918.32 | -4.96e4.00±2.93e4.00 | -4.96e4.00±2.93e4.00 | -4.96e4.00±2.93e4.00 | -4.96e4.00±2.93e4.00 | -781.52±4.39 | -5.20e4.00±3.36e4.00 | -5.20e4.00±3.36e4.00 | -5.20e4.00±3.36e4.00 | -5.20e4.00±3.36e4.00 |
| PRIME | -781.52±4.39 | -5.97±0.85 | -5.97±0.85 | -5.97±0.85 | -5.97±0.85 | -781.52±4.39 | -6.00±0.86 | -6.00±0.86 | -6.00±0.86 | -6.00±0.86 | -781.52±4.39 | -6.01±0.86 | -6.01±0.86 | -6.01±0.86 | -6.01±0.86 |

