# OpenReview forum: "SOO-Bench: Benchmarks for Evaluating the Stability of Offline Black-Box Optimization"
_ICLR.cc/2025/Conference — ICLR 2025 Poster_

### Official Review · Reviewer_8L7a · 2024-10-29

**Soundness:** 2
**Presentation:** 2
**Contribution:** 2
**Rating:** 5
**Confidence:** 4

**Summary:**

This paper proposes SOO-Bench, a benchmark suite to promote stable offline black-box optimization. Authors integrate several real offline optimization tasks and provide offline optimization tasks with different customizable difficulties by narrowing the dataset distribution. Besides, authors introduce an indicator called Stability Improvement (SI) to quantify the stability of the algorithms and conduct experiments to evaluate algorithmic stability and optimality.

**Strengths:**

1. This paper is claimed to be the first work to disclose stability as one of the core objectives in offline black-box optimization.

2. The paper integrates multiple real-world tasks with a unified API.

3. In the experiments, authors conduct comprehensive evaluation on state-of-the-art baselines and provide thoughful insights.

**Weaknesses:**

1. Narrowing data distribution by removing top and bottom solutions cannot address the uneven cover in real-world task dataset. As mentioned in the paper, personal biases, preferences for certain parameter settings, or prior knowledge might lead to uneven sampling in solution space, not the fitness space. Removing top and bottom solutions cannot reflect such situations since the solutions in the middle may still evenly cover the solution space, and the solutions sampled unevenly may also cover both top and bottom solutions. Therefore, narrowing data distribution by removing top and bottom solutions cannot address the uneven cover in real-world task dataset, perhaps narrowing the distribution in solution space could be better.

2. The Stability Improvement indicator proposed in this paper is imperfect in measuring the algorithm stability. According to Equation (2), the way to obtain highest SI is to achieve a solution better than the best solution in the offline dataset and at the first step and retain it till the end. In this case the SI can be 1, but such precocious behavior can lead to poor performance (i.e., COMMs in the experiments has high SI and poor performance). If we consider SI as an indicator only about stability and has nothing to do with performance, another case where the performance keeps increasing and achieves the best performance at the last step (i.e., Algorithm A's curve before achieving objective value a in Figure 2(b)), its SI can be lower than some unstable algorithms. However, we know that in this case, it is a stable algorithm with stable performance improvement along the optimization. In a word, the proposed SI cannot reflect the algorithm stability.

**Questions:**

See weaknesses part.

---

> ### Author Response · Authors · 2024-11-18
> **Reply to Reviewer 8L7a （1/2）**
>
> Thank you for your efforts in providing valuable comments on our work. We appreciate the reviewer's comment about the points that should be further considered from the aspect of our experimental point of view. We have thoroughly considered the concerns raised by the reviewer and revised our manuscript as follows (we also recommend the reviewer check the global response).
>
> ***List of changes in the manuscript***:
>
> > 1. Appendix D are revised to add additional experiments according to Q1.
>
> > **Q1** About experiment setting.
>
> **A1** Thank you for your valuable suggestion. In response, we have narrowed down different regions of the solution space, and the experimental results can be found in Appendix D.2, highlighted in blue. However, we would like to emphasize that **the goal of offline optimization is to stably surpass the offline dataset during optimization procedure**. When there is a significant lack of high-quality $y$ values, the algorithm often faces a serious out-of-distribution generalization problem, which is precisely the challenge that previous offline optimization algorithms have aimed to address. For this reason, we chose to process the y values in our approach, similar to how the first offline optimization benchmark "design-bench" [1] also processed $y$. The different solution distribution results in GTOPX unconstrained scenario can be found in Table 1.
>
> **Table 1**: Overall the different solution distribution results in GTOPX unconstrained scenario. Results are averaged over five times, and “$\pm$” indicates the standard deviation. $f(x_{OFF})$ means the optimal objective function value in the offline dataset. FS (i.e., final score) means the function value that an offline optimization algorithm finds in the final step during optimization process. FS measures optimality while SI measures stability.
>
> | Low-high          | 0-50                    |                 | 10-60                    |                 | 20-70              |                 | 25-75               |                 |
> | ----------------- | ----------------------- | --------------- | ------------------------ | --------------- | ------------------ | --------------- | ------------------- | --------------- |
> | $f(x_{OFF})$      | 45.67                   |                 | 40.12                    |                 | 38.53              |                 | 40.62               |                 |
> | Metric            | **FS ↓**                | **SI ↑**        | **FS ↓**                 | **SI ↑**        | **FS ↓**           | **SI ↑**        | **FS ↓**            | **SI ↑**        |
> | **CMAES**         | 107.89 $\pm$ 150.49     | $-\infty$       | 46.92 $\pm$ 19.31        | $-\infty$       | 47.41 $\pm$ 23.79  | $-\infty$       | 59.78 $\pm$ 49.68   | $-\infty$       |
> | **Tri-mentoring** | **45.67 $\pm$ 4.63**        | $-\infty$       | **40.12 $\pm$ 2.97**         | $-\infty$       | **38.53 $\pm$ 2.48**   | $-\infty$       | 40.62 $\pm$ 1.61    | $-\infty$       |
> | **CBAS**          | **45.67 $\pm$ 4.63**         | 0.00 $\pm$ 0.00 | **40.12 $\pm$ 2.97**         | 0.00 $\pm$ 0.00 | **38.53 $\pm$ 2.48**   | 0.00 $\pm$ 0.00 | **40.62 $\pm$ 1.62**    | 0.00 $\pm$ 0.00 |
> | **ARCOO**         | **45.67 $\pm$ 4.63**         | $-\infty$       | 41.35 $\pm$ 3.00         | $-\infty$       | **38.53 $\pm$ 2.48**   | $-\infty$       | 41.63 $\pm$ 2.81    | $-\infty$       |
> | **TT-DDEA**       | 56527.15 $\pm$ 76221.35 | $-\infty$       | 50225.22 $\pm$ 112867.53 | $-\infty$       | 187.73 $\pm$ 56.53 | $-\infty$       | 549.99 $\pm$ 480.58 | $-\infty$       |
> | **BO**            | 70.88 $\pm$ 8.96        | $-\infty$       | 70.98 $\pm$ 5.78         | $-\infty$       | 82.18 $\pm$ 16.69  | $-\infty$       | 86.16 $\pm$ 17.65   | $-\infty$       |
> | **CCDDEA**        | 53.34 $\pm$ 21.36       | $-\infty$       | **40.12 $\pm$ 2.97**         | 0.00 $\pm$ 0.00 | **38.53 $\pm$ 2.48**   | 0.00 $\pm$ 0.00 | 113.09 $\pm$ 146.09 | $-\infty$       |
>
> The experimental results in Table 1 show that the FS values of all algorithms are close to the optimal solutions of the offline dataset, indicating that under this experimental setting, it is challenging for the algorithms to further improve upon the current solutions. The main reason why FS is the same but SI is different is that there is a difference in the decimals after FS, although rounding is omitted. This phenomenon can be attributed to two key factors:

---

> ### Author Response · Authors · 2024-11-18
> **Reply to Reviewer 8L7a （2/2）**
>
> - Lack of solution space exacerbating the OOD problem: In the experiments, parts of the solution space may not be covered by the offline dataset, which restricts the algorithm’s exploration capabilities during the optimization process. As a result, the algorithm struggles to identify potential optimal solutions within the solution space. This limitation often leads to the "OOD" problem, where the algorithm’s performance in new optimization tasks is constrained by the incomplete representation of the solution space.
> - The offline dataset’s optimal solutions are already highly competitive: The offline dataset itself contains solutions that are close to the global optimum. With such high-quality data, the algorithms have limited room for further optimization, making it difficult to achieve solutions that significantly surpass the current optimal results.
>
> > **Q2** About SI metric.
>
> **A2** Thank you for your insightful feedback. FS represents the optimal solution found by the algorithm after the optimization is completed. This indicator is intended to indicate optimality.
>
> **SI quantifies the stability of an algorithm by comparing its performance to two reference points**: an extremely conservative baseline and an ideal algorithm. The conservative baseline represents the scenario where only the optimal offline dataset is obtained, while the ideal algorithm achieves both perfect optimization and stability returning the best possible solution. The SI value is calculated as the difference between the performance of the given algorithm and the conservative baseline, divided by the difference between the ideal algorithm and the conservative baseline. This metric effectively captures the algorithm's stability throughout the optimization process.
>
> Regarding the issue of precocious behavior, we believe that **both FS and SI should be considered comprehensively**. We do not underestimate the importance of FS. As noted in line 460, when SI is high but FS is low, we do not consider the algorithm to be performing well. Ideally, we aim for future algorithms to achieve stability throughout the entire optimization process while maintaining high FS for maximization problems—this would result in both high SI and high FS.
>
> Regarding stability, we would like to clarify that **we typically set a sufficiently large step size to avoid the issue you mentioned**. Additionally, we want to emphasize that many previous offline optimization algorithms experience performance degradation as optimization progresses, what we don't want to see. Our proposed indicators SI can reveal this phenomenon.
>
> Even in the situation you described, where SI may be lower, the FS might still be relatively high, and this scenario would be acceptable because our goal is to achieve the maximum performance while ensuring stable performance over time. **It is important to clarify that in maximization tasks, algorithms with high FS and SI values are considered advantageous. Conversely, in minimization tasks, algorithms with low FS and high SI values are deemed more favorable.**
>
> [1] Trabucco B, Geng X, Kumar A, et al. Design-bench: Benchmarks for data-driven offline model-based optimization, International Conference on Machine Learning, 2022.

---

> ### Author Response · Authors · 2024-11-25
> **Gentle Reminder of the Revision Deadline**
>
> Dear Reviewer 8L7a,
>
> We express our sincere gratitude for your invaluable comments and constructive feedback on our paper. We understand that you have a busy schedule, and we kindly remind you that the revision deadline is approaching. During the rebuttal period, we diligently addressed your concerns by providing point-to-point responses, which included **the incorporation of new experiments (such as the different solution distributions)** , and **the clarification on the expression of SI metric**. If you have any further questions or concerns, please do not hesitate to let us know. Moreover, if you find our response satisfactory, could you please kindly consider the possibility of updating the rating. Thank you very much for your valuable suggestion.
>
> Thank you for your attention to our work, and we look forward to your response.
>
> Best regards,
>
> The Authors.

---

> > ### Comment · Reviewer_8L7a · 2024-11-25
> >
> > I appreciate the authors' detailed responses.
> >
> > A1: I cannot find the Appendix D.2, but I see the results in Appendix D, in which authors present the results of different solution space distributions. However, firstly, how to generate these solution distributions is omitted. The meaning of "Low-high" values in the table is not explained either. Therefore the conclusions are not convincing. Besides, although the lack of high-quality y values can be a challenge, I don't think constructing the offline dataset by processing the y values only is a good idea. As the paper claims in lines 051-053, the data distributions may be narrowed by experimenters' subjective opinions in the solution space. In practice, we cannot construct a dataset by processing the y values and obtain a dataset with y values in a particular range. In my opinion, constructing dataset in solution space may be more convincing.
> >
> > A2: Firstly, as an indicator, SI should be able to evaluate algorithms independently. For example, algorithm *A*'s SI is larger than *B*'s, then *A* should be considered to be more stable than *B*. However, as I mentioned in Q2, SI cannot indentify the algorithms with increasing performance. If SI can only be a supplement of FS, the contribution is limited. Secondly, authors claim that they use sufficiently large step size to avoid the issue, it means that the effectiveness of SI relies on the step size, which should not be considered as a factor of optimization stability. It could be a limitation since in practice we may not ensure sufficiently steps (i.e., limited computational resources or low-performance computing platforms), in these cases SI may fail in avoiding the issue. Besides, it is an important question that how many steps can be considered as "sufficient", is it the same across all datasets and all algorithms? The boundary of "sufficient" is blurry.
> >
> > Therefore, I maintain my original rating of 3.

---

> ### Author Response · Authors · 2024-11-28
> **Reply to Reviewer 8L7a （1/4）**
>
> Thank you very much for your careful review and feedback!
>
> > Q1: About solution space distribution.
>
> **A1**: Thank you for your valuable comments. Black-box optimization has been developed and applied across a wide range of fields. However, its widespread adoption in real-world applications has been hindered by the need for frequent evaluations. In many tasks, evaluating the objective function is costly, risky, or even infeasible. Although active evaluations of new solutions may not always be feasible, historical data in the form of solution-function value pairs is often available. Offline optimization, which does not require real-time querying of solutions, aims to **leverage this offline data to recommend high-quality solutions for real-world applications**. In many domains, such as experimental chemistry, researchers cannot explore the entire solution space. Instead, they typically make slight adjustments based on prior experiments, meaning the solution (denoted as $x$) may remain similar, while the objective value $y$ changes. One of the primary challenges in offline optimization is that this optimization process carries a risk of performance degradation due to the overestimation error in the surrogate model [2, 3, 4, 5]. A well-trained surrogate model can provide accurate predictions for solutions near the offline data, but it may **make significant errors when predicting out-of-distribution (OOD) solutions** [2, 3, 4, 5]. As a result, optimization algorithms tend to explore unseen OOD solutions in the hope of discovering better-performing solutions (i.e., solutions that outperform the best ones in the offline dataset). However, as the optimization progresses, the risk of performance degradation increases, and the quality of the found solutions may decline. Without active evaluation feedback, it becomes difficult to determine when to stop the optimization and return the final solution. Therefore, the core goal of offline optimization is to **stably surpass the offline dataset during optimization procedure**.
>
> That is to say, the goal is to identify the best possible solutions (i.e., those with high $y$-values) based on historical data. This goal cannot be effectively achieved by simply adjusting the distribution of the solution space. As noted in Line 37 of the paper, the objective function is typically a black-box function, meaning we lack prior knowledge of which regions contain the global optimum. **Excessive exploration of the solution space would not only be resource-intensive but may also fail to identify the global optimum**. Additionally, uniform sampling of the solution space could inadvertently cover regions containing the global optimum. If regions near the global optimum are included in the dataset, offline optimization algorithms, which are proposed to solve OOD issue, might fail to distinguish between different solutions.
>
> While obtaining the global optimum may not always be feasible, we can identify regions with relatively high $y$-values and exclude these areas from the dataset. This ensures that the dataset avoids regions near the global optimum, which is essential for constructing a more reliable offline dataset tailored for offline optimization.
>
> Therefore, we believe it is more reasonable to process the dataset based on $y$-values, similar with famous benchmark design_bench in offline optimization [1]. While the approach we propose is relatively simple, it effectively addresses the distribution issue. We also provide guidance on how to adaptively construct a dataset on GitHub [link], where users can customize their datasets.

---

> ### Author Response · Authors · 2024-11-28
> **Reply to Reviewer 8L7a (2/4)**
>
> > Q2: About SI metric.
>
> **A2**: Thanks for your constructive suggestions. First and foremost, we would like to thank the reviewer for their valuable feedback. Initially, our focus was primarily on algorithm stability, an aspect that is often overlooked in existing research. Our original approach was to evaluate algorithm stability in conjunction with FS to provide a more comprehensive assessment of algorithm performance. After considering the reviewer's suggestions, we realized the importance of accounting for performance improvement, and we have since revised our approach to better address the identification of increasing performance. In the revised version, we have incorporated this idea to ensure that new metric better reflects performance improvement in various contexts.
>
> We greatly appreciate the reviewer’s insight, which has been instrumental in refining our work and making significant improvements to the paper.
>
> Specifically, to provide more direct representations in each aspect, we further propose two independent indicators to represent the optimality and stability, respectively.
>
> **Optimality Indicator (OI)** compares a given algorithm $A$ to a hypothetical algorithm $A_d$ that returns the best solution in offline static data in every step $t$. The algorithm $A_d$ is considered as a fundamental baseline that is extremely conservative and any effective algorithm is expected to surpass it. Hence, OI indicates the optimality of $A$ by showing how well it outperforms $A_d$. Formally the OI of $A$ is calculated as $OI(A) = \frac{S_A}{S{A_d}}$ , where $S_A$ represents the shaded area under the curve of $A$ and $S{A_d}$ represents the shaded area under the curve of $A_d$. **The OI indicates the optimality of algorithm $A$ by showing how well it outperforms the hypothetical baseline algorithm $A_d$**.
>
> **Stability Indicator (SI)** compares $A$ to another hypothetical algorithm $A_o$ that always returns the best solution encountered during the entire optimization procedure of $A$ at every step t. The algorithm $A_o$ is considered as an ideal algorithm that achieves both the best optimality and stability (i.e., consistently returns the best solution). Therefore, **SI indicates the stability of $A$ by measuring how close its solution scores are to those of the ideal algorithm $A_o$**. A higher SI value implies that in more steps, the scores of the solutions generated by algorithm $A$ are closer to the best solution, indicating higher stability. Conversely, a lower SI value suggests greater variability in the solution scores and lower stability. Formally the SI of $A$ is calculated as $SI(A) = \frac{S_A}{S_{A_o}}$ , where $S{A_o}$ represents the shaded area under the curve of the hypothetical ideal algorithm $A_o$.
>
> Thus, we introduce a novel metric, **Stability-Optimality metric (SO)** in Definition 1, for offline MBO to assess the model's capability in identifying the stable and optimal one from a set of candidate designs in the whole optimization process.
>
> **Definition 1 (SO metric for Offline MBO)**: For each optimization step $t \in \lbrace \{1, 2, \cdots, T \rbrace \}$,
> $$
> w(t) = \max(0, 1-\frac{t}{T+1}).
> $$
> The function $w(t)$ decreases gradually over time. In the early stages, greater emphasis is placed on achieving optimality, while in the later stages, the focus transitions toward stability. Inspired by F1 score, the SO metric is defined as:
> $$
> SO=\frac{2\cdot OI\cdot SI}{w(t)\cdot SI + (1-w(t))\cdot OI}.
> $$
> **The SO metric for offline MBO can effectively evaluate a model's ability about stability and optimality**. A higher SO value indicates better performance in optimality and stability. In the early stages of optimization, the focus is on achieving optimality, as we aim to make significant improvements during this phase. As the optimization progresses and reaches later stages, the priority shifts toward maintaining stability, ensuring that performance does not degrade significantly due to out-of-distribution generalization, even after extensive optimization.

---

> ### Author Response · Authors · 2024-11-28
> **Reply to Reviewer 8L7a (3/4)**
>
> To better clarify, let’s consider the following three sequences:
>
> A: [0.4, 0.5, 0.5, 0.5, 0.5, 0.5, 0.5, 0.5, 0.5, 0.5]
>
> B: [0.4, 0.5, 0.5, 0.6, 0.6, 0.7, 0.7, 0.7, 0.8, 0.8]
>
> C: [0.4, 0.8, 0.8, 0.8, 0.8, 0.8, 0.8, 0.8, 0.8, 0.8]
>
> As you pointed out, in our previous indicator SI, we considered sequences A and C to be very stable, which led to the same SI value of 0.944 for both. However, we did not account for the actual FS value. In contrast, for sequence B, which is steadily increasing, an SI value of 0.583 at this stage. The new indicator we proposed addresses all the issues you raised previously, balancing both stability and optimality. For the A, B, and C sequences discussed earlier, assuming the total step length $T=200$, consistent with our previous settings, the SO values at this stage are 2.442, 3.017, and 3.705, respectively. This outcome aligns with our intuitive expectations. **In summary, Sequence B represents the algorithm with the increasing performance you mentioned. This is the issue you pointed out earlier, which SI can not address. Our newly improved metric SO effectively resolves this problem. Additionally, the precocious behavior you mentioned can lead to poor performance, as seen with Sequence A, which receives a low value with the new metric SO.** We are confident that our improved indicator addresses all of your concerns.
>
> We agree that the new metric SO can validate offline optimization approachs on our benchmark with diverse backgrounds strengthens the paper's credibility. Here we only show the GTOPX unconstrained scenario. To further simulate a narrow distribution, the experimental results for the cases where $m=25$% is missing near the worst value and $n=75$% is missing near the optimal value are as follows:
>
> **Table 1: Overall results on GTOPX unconstrained scenario. Results are averaged over five runs. $f(x_{OFF})$ represents the optimal objective function value in the offline dataset. FS means the function value that an offline optimization algorithm finds in the final step during the optimization process. SI is our previous metric, while SO, our new metric, evaluates both optimality and stability.**
>
> | Benchmark （FS / SI / SO)     | GTOPX 2        | GTOPX 3         | GTOPX 4         | GTOPX 6         |
> |---------------|-------------------|--------------------|-------------------|--------------------|
> | $f(x_{OFF})$ | 196.21            | 151.68             | 216.34            | 112.11              |
> | **BO**        | 95.45$\pm$18.43 / 0.69$\pm$0.02 / 1.09$\pm$0.05 | 75.78$\pm$29.38 / 0.70$\pm$0.03 / 0.97$\pm$0.05 | 117.39$\pm$17.11 / 0.68$\pm$0.04 / 1.17$\pm$0.09 |  58.11$\pm$8.25 / 0.63$\pm$0.04 / 1.13$\pm$0.07 |
> | **CMAES**     | 196.21$\pm$1.18 / 0.00$\pm$0.00 / 1.82$\pm$0.14 | 151.68$\pm$0.53 /  0.00$\pm$0.00 / 1.95$\pm$0.04 |  216.34$\pm$0.61 / 0.00$\pm$0.00 / 2.17$\pm$0.78 |  112.11$\pm$0.38 / 0.00$\pm$0.00 / 1.94$\pm$0.09 |
> | **CBAS**      | 196.21$\pm$1.18 / 0.00$\pm$0.00 / 2.00 $\pm$ 0.00 | 86.51$\pm$3.97 / 0.71$\pm$0.06 / 1.30$\pm$0.05 | 208.80$\pm$15.50 / 0.14$\pm$0.02 / 2.00$\pm$0.00 |  87.08$\pm$31.05 / 0.34$\pm$0.09 / 1.60$\pm$0.07 |
> | **TTDDEA**    |  224.17$\pm$53.87 / $-\infty$ / 8.52$\pm$9.58 | 156.92$\pm$91.05 / $-\infty$ / 6.89$\pm$3.50 |  260.4$\pm$54.89 / $-\infty$ / 9.52$\pm$8.64 | 148.76$\pm$50.67 / $-\infty$ / 2.31$\pm$0.64 |
> | **ARCOO**     |  90.73$\pm$10.98 / 0.78$\pm$0.05 / **0.85$\pm$0.06** |  65.88$\pm$13.12 / 0.85$\pm$0.01 / **0.77$\pm$0.06** | 102.84$\pm$21.76 / 0.79$\pm$0.04 / **1.01$\pm$0.10** | 65.17$\pm$13.30 / 0.74$\pm$0.08 / **1.01$\pm$0.09** |
> | **Tri-mentoring** | 129.47$\pm$54.75 / $-\infty$ / 1.82$\pm$0.41 | 140.23$\pm$22.88 / $-\infty$ / 1.67$\pm$0.32 | 176.31$\pm$37.30 / 0.86$\pm$0.18 / 1.77$\pm$0.36 |  112.11$\pm$0.38 / $-\infty$ / 1.76 ± 0.37 |
> | **CCDDEA**    | 197.25$\pm$3.70 / $-\infty$ / 1.98$\pm$0.04 | 152.71$\pm$4.01 / $-\infty$ / 2.03$\pm$0.08 |  216.18$\pm$3.29 / $-\infty$ / 2.03$\pm$0.05 | 112.46$\pm$1.88 / $-\infty$ / 2.06$\pm$0.07 |
>
> By comparing the results of FS and SI , we found that our new metric SO can take both optimality and stability into consideration. Since GTOPX unconstrained tasks are minimization tasks, SO should be as small as possible. we observed the advantages of the ARCOO algorithm, which can dynamically control the optimization risk during the optimization process. Existing offline BBO methods (e.g., CBAS, ARCOO) perform well across different tasks, except for TTDDEA. This is because TTDDEA, being an evolutionary-based method, relies on high-confidence data augmentation without considering the out-of-distribution issue. All these experimental results validate the importance of steady improvement during the optimization process.
>
> Due to time constraints, we do not have time to revise the full text. We will revise the experiments and expressions of the full text later. Thank you for your reply.

---

> ### Author Response · Authors · 2024-11-30
> **Reply to Reviewer 8L7a (4/4)**
>
> > Q3: About typos.
>
> **A3**: Thanks for pointing these out. About how to generate these solution distributions and the meaning of "Low-high" values, we have explained them in Line 1536-1546. "Low-high" have been modified to "Left-Right”, which means right-$n\%$ of the solution space range and the left-$m\%$ of the solution space range are removed. We have revised them and proofread thoroughly according to your suggestions. Thank you very much.
>
> **About how to generate these solution distributions**: First, we provide a black-box ground-truth oracle objective function for each task. An initial dataset is obtained by uniformly sampling and evaluating the objective function. Datasets of different difficulty levels are constructed based on this sorted initial dataset. Specifically, the right-$n\%$ of the solution space range and the left-$m\%$ of the solution space range are removed to show different solution space distributions. Through the above steps, an offline dataset with a narrow distribution in real tasks is constructed by removing solutions.
>
> To simulate a more realistic data distribution, we choose a dataset size that is 1000 times the variable dimension. At the same time, to further simulate the narrow distribution, missing the $m\%$ and the $n\%$ are used to construct different solution distributions. In this paper, we select the middle 50\% of the data (i.e., $m\%-n\%=50\%$) to construct a simulated dataset as a reasonable baseline. Since the proposed benchmark is highly flexible and customizable, it enables users to modify the data volume, $m\%$, and $n\%$ as needed.
>
> [1] Trabucco B, Geng X, Kumar A, et al. Design-bench: Benchmarks for data-driven offline model-based optimization, International Conference on Machine Learning, 2022.
>
> [2] Han Qi, Yi Su, Aviral Kumar, and Sergey Levine. Data-driven model-based optimization via invariant representation learning, in Advances in Neural Information Processing Systems 35, 2022.
>
> [3] Brandon Trabucco, Aviral Kumar, Xinyang Geng, and Sergey Levine,. Conservative objective models for effective offline model-based optimization, in Proceedings of the 38th International Conference on Machine Learning, 2021.
>
> [4] Ye Yuan, Can (Sam) Chen, Zixuan Liu, Willie Neiswanger, and Xue (Steve) Liu. Importance-aware co-teaching for offline model-based optimization. In Advances in Neural Information Processing Systems 36, 2023.
>
> [5] Taeyoung Yun, Sujin Yun, Jaewoo Lee, and Jinkyoo Park. Guided trajectory generation with diffusion models for offline model-based optimization. In Advances in Neural Information Processing Systems 37, 2024.
>
> **If there are any issues, please feel free to reply, and we will respond promptly.**

---

> ### Author Response · Authors · 2024-12-03
> **Kind Reminder**
>
> Dear Reviewer 8L7a,
>
> As the deadline approaches, we sincerely hope to address your concerns and discuss the rebuttal with you further. If you have any questions, please feel free to ask directly. Moreover, if you find our response satisfactory, could you please kindly consider the possibility of updating the rating. Thank you very much for your valuable suggestion.
>
> Best regards,
>
> The authors

---

> > ### Comment · Reviewer_8L7a · 2024-12-03
> >
> > Thanks for the authors' thoughtful responses and detailed experiments. For A1, I agree that it is reasonable to use y values for those datasets without available global optimums or known feasible solution spaces. For A2, the modified indicator address my concern, which relieves the dependency on optimization steps and reflects the performance improvement along the optimization. Since my concerns are addressed, I raise my score.

---

> > > ### Author Response · Authors · 2024-12-03
> > > **Official Comment by Authors**
> > >
> > > Thank you for your prompt and positive feedback, as well as for appreciating the value of our work and raising the score. We are delighted to have addressed your questions and incorporated all your suggested modifications into the revised version based on your comments. If you have any further questions or concerns, please don’t hesitate to reach out.

---

### Official Review · Reviewer_9Mjy · 2024-10-31

**Soundness:** 4
**Presentation:** 3
**Contribution:** 3
**Rating:** 8
**Confidence:** 5

**Summary:**

The authors present an innovative benchmark for evaluating the stability of offline data-driven optimization problems. They employ a customized data distribution approach that removes points near optimal and suboptimal values, and introduce a Stability Improvement (SI) indicator to measure stability. Their evaluation compares various offline optimization algorithms — six for unconstrained problems, plus four for constrained ones. The results reveal diverse performance levels across the benchmark's instances, showcasing its effectiveness in assessing stability.

**Strengths:**

- The paper provides key insights into the stability of offline optimization methods, advancing this emerging field.
- It is clearly written and well-structured.
- The related work section is thorough, and the introduction effectively contextualizes the research.
- The benchmark is detailed, with clear explanations of its tasks and datasets.
- Open-source availability enhances interpretability and reproducibility, backed by comprehensive experiments.

**Weaknesses:**

Please refer to the Questions section.

**Questions:**

- There appear to be some typos, specifically "TTDEEA" in Table 2 and Section 6.2, and "TTDEA" in Appendix C.

- For the new SI indicator, how should we best assess an algorithm's performance? In which cases would a lower FS be advantageous, and when should we prioritize a higher SI?

- I also have a question regarding the paper’s submission history. I was one of the reviewers for its NeurIPS submission this year, where I saw that all reviewers ultimately gave positive scores. However, I was surprised to see that the authors decided to withdraw the paper. Could you share the reason for this decision? Additionally, I noticed that some specific revisions the authors intended to make during the NeurIPS rebuttal are not reflected in the current ICLR submission.

---

> ### Author Response · Authors · 2024-11-18
> **Reply to Reviewer 9Mjy (1/2)**
>
> We sincerely appreciate your efforts in reviewing our paper. We are pleased that the reviewer noticed the strengths regarding the SOO-Bench. We have carefully reviewed the weaknesses pointed out by the reviewer and have revised our manuscript as follows (we also recommend the reviewer review the overall response).
>
> ***List of changes in the manuscript***:
>
> > 1. The typos are fixed according to Q1.
> > 1. Evaluate new offline optimization algorithm CCDDEA on our benchmark according to Q3.
>
> > **Q1** About typos.
>
> **A1** Thank you for pointing out the typos. We have fixed it in the updated PDF, and the correction is highlighted in blue.
>
> > **Q2** About FS and SI metric.
>
> **A2** Thank you for your questions. In the performance evaluation of offline optimization algorithms, both FS and SI indicators must be considered. In general, for maximization problems, **the goal is to make FS and SI as large as possible**. However, the trade-off between these two indicators can vary depending on the specific problem. For example, in some tasks, the focus may not be on finding the highest possible solution but rather on ensuring the stability of the solution. In such cases, more attention should be given to the SI indicator. This process is akin to finding the Pareto optimum between the FS and SI indicators, where different preferences will lead to different outcomes. But we want to restate that high FS and SI are the ultimate goal in the maximization problems.
>
> > **Q3** About submission history.
>
> **A3** Thank you for your question regarding the paper’s submission history. Indeed, the NeurIPS score was high, with three positive reviews and one borderline negative review. However, the negative review did not adequately involve in the rebuttal discussion, which ultimately led to the Area Chair's rejection. This outcome took us by surprise. We decided to withdraw the paper only after the rejection. Given the short time between the rejection and the ICLR submission, we didn’t have enough time to incorporate the additional experiments mentioned in the NeurIPS rebuttal. We have now included the relevant experiments in the Section 6.1, Table 2, 5, 9, Appendix B.1 only for GTOPX tasks. We are making further improvements to the manuscript for other tasks now. The CCDDEA results in GTOPX unconstrianed scenario can be found in Table 1.
>
> **Table 1**: Overall CCDDEA results in GTOPX unconstrianed scenario. The distribution m-n means the missing m% near the
> worst value and the missing n% near the optimal value. Results are averaged over five times, and “$\pm$” indicates the standard deviation. $f(x_{OFF})$ means the optimal objective function value in the offline dataset. FS (i.e., final score) means the function value that an offline optimization algorithm finds in the final step during optimization process. FS measures optimality while SI measures stability.
>
> | Tasks     | GTOPX 2           |           | GTOPX 3             |           | GTOPX 4           |           | GTOPX 6            |           |
> | --------- | ----------------- | --------- | ------------------- | --------- | ----------------- | --------- | ------------------ | --------- |
> |   m-n | **FS ↓**          | **SI ↑**  | **FS ↓**            | **SI ↑**  | **FS ↓**          | **SI ↑**  | **FS ↓**           | **SI ↑**  |
> | **0-60**  | 276.57 $\pm$ 4.77 | $-\infty$ | 228.19 $\pm$ 8.06   | $-\infty$ | 326.68 $\pm$ 5.93 | $-\infty$ | 142.58 $\pm$ 12.14 | $-\infty$ |
> | **10-60** | 276.57 $\pm$ 4.77 | $-\infty$ | 228.19 $\pm$ 8.06   | $-\infty$ | 326.68 $\pm$ 5.93 | $-\infty$ | 155.44 $\pm$ 33.21 | $-\infty$ |
> | **0-70**  | 219.22 $\pm$ 3.95 | $-\infty$ | 172.37 $\pm$ 5.26   | $-\infty$ | 242.95 $\pm$ 4.04 | $-\infty$ | 121.70 $\pm$ 1.72  | $-\infty$ |
> | **20-70** | 219.22 $\pm$ 3.95 | $-\infty$ | 172.36 $\pm$ 5.27   | $-\infty$ | 242.95 $\pm$ 4.04 | $-\infty$ | 121.70 $\pm$ 1.72  | $-\infty$ |
> | **25-75** | 197.25 $\pm$ 3.70 | $-\infty$ | 152.71 $\pm$ 4.01   | $-\infty$ | 216.18 $\pm$ 3.29 | $-\infty$ | 112.46 $\pm$ 1.88  | $-\infty$ |
> | **30-80** | 176.53 $\pm$ 3.76 | $-\infty$ | 135.21 $\pm$ 2.74   | $-\infty$ | 193.12 $\pm$ 2.67 | $-\infty$ | 116.20 $\pm$ 33.92 | $-\infty$ |
> | **0-80**  | 176.53 $\pm$ 3.76 | $-\infty$ | 130.06 $\pm$ 15.23  | $-\infty$ | 197.42 $\pm$ 10.85 | $-\infty$ | 103.49 $\pm$ 1.53  | $-\infty$ |
> | **40-90** | 138.24 $\pm$ 2.59 | $-\infty$ | 102.86 $\pm$ 2.54   | $-\infty$ | 153.43 $\pm$ 3.53 | $-\infty$ | 83.20 $\pm$ 1.05   | $-\infty$ |
> | **0-90**  | 138.24 $\pm$ 2.59 | $-\infty$ | 491.55 $\pm$ 954.25 | $-\infty$ | 153.44 $\pm$ 3.53 | $-\infty$ | 83.20 $\pm$ 1.05   | $-\infty$ |
>
> As shown in Table 1, although the SI of CCDDEA is relatively poor, this is mainly because the task is difficult and it is easy to overestimate when exploring the OOD area. However, compared with other evolutionary algorithms such as TTDDEA, it is found that CCDDEA can be closer to the optimal solution.

---

> ### Author Response · Authors · 2024-11-18
> **Reply to Reviewer 9Mjy (2/2)**
>
> Table 2 and 3 shows performance of CCDDEA in the different size of the offline dataset and the different solution distribution. They are proposed by other reviewers of this ICLR paper. We also use CCDDEA as the baseline and put the method description and pseudocode of CCDDEA in Appendix B. We found that even in complex settings, such as changing the amount of data or changing the distribution of the solution space, CCDDEA is still a relatively competitive method in terms of FS indicators.
>
> **Table 2**: Overall results for different sizes of the offline dataset using the CCDDEA method in the GTOPX unconstrained scenario. Details are the same as Table 1.
>
> | Datasize          | N=dim*100          |             | N=dim*300      |             | N=dim*500      |              |
> | ----------------- | ------------------ | ----------- | -------------- | ----------- | -------------- | ------------ |
> | **$f(x_{OFF})$**       | 197.25                 |                 | 195.40             |                 | 195.66             |                  |
> | Metrics           | **FS ↓**               | **SI ↑**        | **FS ↓**           | **SI ↑**        | **FS ↓**           | **SI ↑**         |
> | **CCDDEA**        | 197.25 $\pm$ 3.70      | 0.00 $\pm$ 0.00 | 195.40 $\pm$ 2.62  | 0.00 $\pm$ 0.00 | 167.82 $\pm$ 44.25 | -0.02 $\pm$ 0.11 |
>
> **Table 3**: Overall results for the different solution distribution using the CCDDEA method in the GTOPX unconstrained scenario. Details are the same as Table 1.
>
> | Low-high          | 0-50                    |                 | 10-60                    |                 | 20-70              |                 | 25-75               |                 |
> | ----------------- | ----------------------- | --------------- | ------------------------ | --------------- | ------------------ | --------------- | ------------------- | --------------- |
> | $f(x_{OFF})$      | 45.67                   |                 | 40.12                    |                 | 38.53              |                 | 40.62               |                 |
> | Metric            | **FS ↓**                | **SI ↑**        | **FS ↓**                 | **SI ↑**        | **FS ↓**           | **SI ↑**        | **FS ↓**            | **SI ↑**        |
> | **CCDDEA**        | 53.34 $\pm$ 21.36       | $-\infty$       | 40.12 $\pm$ 2.97         | 0.00 $\pm$ 0.00 | 38.53 $\pm$ 2.48   | 0.00 $\pm$ 0.00 | 113.09 $\pm$ 146.09 | $-\infty$       |

---

> > ### Comment · Reviewer_9Mjy · 2024-11-19
> >
> > Thank you to the authors for the detailed and thoughtful response. All of my concerns have been addressed, and I am happy to raise my score.

---

> > > ### Author Response · Authors · 2024-11-19
> > > **Thanks for your response**
> > >
> > > We sincerely thank you for your response and for raising the score. Your valuable suggestions are important in helping us improve our work. We are truly grateful for your time and effort in reviewing our manuscript. Please let us know if you have any further questions.

---

### Official Review · Reviewer_Lkg5 · 2024-11-03

**Soundness:** 3
**Presentation:** 4
**Contribution:** 3
**Rating:** 5
**Confidence:** 5

**Summary:**

This paper proposes a new benchmark suite for offline optimization. The two key contributions are (1) the ability to adjust the left-out portion (top m% and bottom n%) in the construction of the offline dataset; and (2) a new evaluation metric called SI that measures the stability of the recommendations. The paper shows that existing methods have varying performances when adjusting the dataset difficulty via controlling the parameters m and n. Several experiments also reveal the unstable behaviors of previous methods. Finally, compared to the Design-Bench (which is the default benchmark for offline opt until recently), the proposed benchmark also offers several new constrained optimization tasks.

**Strengths:**

The paper points out some clear weaknesses of the Design-Bench and proposes practical solutions to fix them. Empirical results point out some interesting insights and fill in missing gaps in current offline opt research. The paper is also clearly written and well-motivated.

**Weaknesses:**

- I get the point of the SI metric, but don't quite agree with its formulation. Other than the thresholding with respect to the best offline value, how is it fundamentally different from computing variance? The way I see it, even if we incorporate the best offline value into the formulation, the SI metric alone is not enough to reflect performance. For example, let's say the best offline value is 0, a recommendation of 0.1, 0.1, 0.1 .... 0.1 will obtain better SI score than -0.1, 10, 10, 10, 10 .... A better way might be to treat the list of recommendations as a distribution, then perform hypothesis testing to check if the mean of this distribution is greater than the best offline value. The p-value would then indicate the performance level. It doesn't seem like there is a need to reinvent the wheel here.

-  One would expect higher difficulty to be achieved by leaving out high values in the offline dataset. This seems to be the case with the FS metric (table 5 in appendix). In the Design-Bench paper I think the bottom 30% of the entire dataset was selected as the offline data, which is likely harder than most of the settings presented. Even at this difficulty, the baseline methods are already quite competitive -- each new paper will report a different set of result (typically indicating that it is best). Personally, I think the focus should be on making the benchmark more challenging without severing limit the size of the offline dataset.

- Additionally, the fluctuating performance of baseline methods suggest that randomness plays a huge factor in offline opt. I would be more interested in designing benchmarks to eliminate this factor.

- Finally, can the authors elaborate on the technical challenge behind creating this baseline? To me, both the new metric and the customizable dataset construction seem like straight-forward wrappers that can be added to the Design-Bench, without remaking it from scratch.

**Questions:**

Please see above

**Details Of Ethics Concerns:**

No ethical concerns.

---

> ### Author Response · Authors · 2024-11-18
> **Reply to Reviewer Lkg5 (1/3)**
>
> Thank you for your efforts in providing valuable comments on our work. We appreciate the reviewer's comment about the points that should be further considered from the aspect of our experimental point of view. We have thoroughly considered the concerns raised by the reviewer and revised our manuscript as follows (we also recommend the reviewer check the global response).
>
> ***List of changes in the manuscript***:
>
> > 1. Appendix D are revised to add results for offline datasets of different sizes according to Q2.
>
> > **Q1** About SI metric.
>
> **A1** Thank you for your questions. We address your questions from the three aspects: the relationship between SI and variance, the limitations of SI in reflecting performance independently, and the issues for evaluating performance directly through p-value.
>
> - **About the relationship between SI and variance**: Using the sequence [0.1,0.5,0.1,−0.3,−0.3,−0.5] as an example, with the best offline dataset value is 0.1, we illustrate their differences. In this case, SI is −0.417. The negative SI value indicates poor stability, as most solutions fail to surpass the offline optimal value in offline dataset. This reflects that the offline optimization algorithm lacks the stability necessary to surpass the offline benchmark effectively.
>
>   By contrast, the variance is relatively small (0.112), indicating that the values are narrowly distributed around the mean. However, this measure does not reveal that most values are below the best offline dataset value, nor does it evaluate whether the algorithm meets the offline optimization goal (i.e., the goal of offline optimization is to stably surpass the offline dataset during optimization procedure).
>
>   This example highlights the difference between SI and variance. Variance measures dispersion but **is agnostic to task-specific objectives**. SI, on the other hand, evaluates stability in the optimization process. This differences emphasize why SI is better suited for stability assessment in offline optimization. All in all, variance is a single evaluation metric, which does not capture the relationship between algorithms' results and the best offline dataset value.
>
> - **About the limitations of SI in reflecting performance independently**: SI measures an algorithm's stability by comparing its performance to two benchmarks: a conservative baseline (optimal offline dataset value) and an ideal algorithm (perfect optimization and stability). This metric effectively captures the algorithm's stability throughout the optimization process. However, offline optimization requires not only stability, it also needs to find the best possible solution, so FS is equally important. **We believe that both FS and SI should be considered comprehensively**. We do not underestimate the importance of FS. As noted in line 460, when SI is high but FS is low, we do not consider the algorithm to be performing well. Ideally, we aim for future algorithms to achieve stability throughout the entire optimization process (i.e., high SI) while maintaining optimality (i.e., high FS for maximization problems and low FS for minimization problems).
>
> - **About the issues for evaluating performance directly through p-value**: While hypothesis testing, can provide statistical insights into stability to a certain extent, it falls short in certain critical aspects. Specifically, hypothesis testing cannot distinguish solutions that are worse than the offline optimal value in offline dataset, similar to the limitations of variance. For example, in cases where most solutions fail to exceed the offline optimal value in offline dataset, a good p-value might suggest good performance, but this ignores the crucial fact that the algorithm generates worse solutions than the offline optimal value in offline dataset. In contrast, the SI metric is explicitly designed to evaluate stability in the context of optimization by comparing the algorithm's cumulative performance to the offline optimal value in offline dataset and the optimal solution found by the algorithm. This ensures that SI reflects both stablity and success in surpassing the offline dataset during optimization procedure.
>
>   In summary, while hypothesis testing can complement stability analysis, it cannot replace SI, as it overlooks the iterative nature of optimization processes, while SI provides a goal-oriented evaluation of algorithm performance.
>
> Finally, we would like to emphasize that while variance and p-value can provide some insight into stability, **they are single evaluation methods**. In contrast, our SI indicator offers a more comprehensive assessment by incorporating both the performance of the optimal solution found by the algorithm and the offline optimal value in offline dataset. This alignment reflects the core goal of offline optimization: to **steadily surpass the offline dataset during optimization procedure**. This makes SI a more comprehensive and meaningful measure compared to variance or p-value.

---

> > ### Comment · Reviewer_Lkg5 · 2024-11-25
> > **Thanks for your response**
> >
> > On your response about the relationship between SI and variance -- I do agree with this explanation. In fact, I did say that "Other than the thresholding with respect to the best offline value ..."
> >
> > On your response about the limitations of SI in reflecting performance independently -- Again I do agree. Both metrics should be considered comprehensively. But if you instead consider FS together with variance, shouldn't that already take care of the "goal-oriented" desiderata? e.g., something with low variance and high FS indicates solutions with high stability well beyond the best offline data point.
> >
> > On your response about the issues for evaluating performance directly through p-value: I don't get this. Can't we do hypothesis testing of the candidate distribution directly against the best offline point?

---

> ### Author Response · Authors · 2024-11-18
> **Reply to Reviewer Lkg5 (2/3)**
>
> > **Q2** About benchmark setting.
>
> **A2** Many thanks for the constructive comments. We would like to highlight that the key difference between our previous offline optimization benchmark “design-bench" is our ability to **adaptively adjust the data distribution** and our **in-depth thinking on the essential goals of offline optimization** (i.e., the goal of offline optimization is to stably surpass the offline dataset during optimization procedure). This feature allows the algorithm to explore performance more thoroughly. In response to your feedback, we have modified the size of the offline dataset. The updated results are shown in blue in Appendix D of the modified version. Refer to Table 1 for results concerning various sizes of the offline dataset in the GTOPX unconstrained scenario.
>
> **Table 1**: Overall the different size of the offline dataset results in GTOPX unconstrained scenario. Results are averaged over five times, and “$\pm$” indicates the standard deviation. $f(x_{OFF})$ means the optimal objective function value in the offline dataset. FS (i.e., final score) means the function value that an offline optimization algorithm finds in the final step during optimization process. FS measures optimality while SI measures stability.
>
> | Datasize          | N=dim*100              |                 | N=dim*300          |                 | N=dim*500          |                  |
> | ----------------- | ---------------------- | --------------- | ------------------ | --------------- | ------------------ | ---------------- |
> | **$f(x_{OFF})$**  | 197.25                 |                 | 195.40             |                 | 195.66             |                  |
> | Metrics           | **FS ↓**               | **SI ↑**        | **FS ↓**           | **SI ↑**        | **FS ↓**           | **SI ↑**         |
> | **CMAES**         | 179.21 $\pm$ 38.49     | 0.03 $\pm$ 0.07 | 157.41 $\pm$ 51.96 | 0.04 $\pm$ 0.06 | 172.74 $\pm$ 40.97 | 0.04 $\pm$ 0.07  |
> | **Tri-mentoring** | 197.25 $\pm$ 3.70      | 0.00 $\pm$ 0.00 | 195.40 $\pm$ 2.62  | $-\infty$       | 179.85 $\pm$ 42.05 | $-\infty$        |
> | **CBAS**          | 197.25 $\pm$ 3.7       | 0.00 $\pm$ 0.00 | 195.40 $\pm$ 2.61  | 0.00 $\pm$ 0.00 | 195.66 $\pm$ 1.78  | 0.00 $\pm$ 0.00  |
> | **ARCOO**         | 99.07 $\pm$ 18.99      | 0.75 $\pm$ 0.07 | 93.22 $\pm$ 23.27  | 0.79 $\pm$ 0.03 | 86.68 $\pm$ 17.22  | 0.78 $\pm$ 0.05  |
> | **TT-DDEA**       | 5594.65 $\pm$ 13897.52 | $-\infty$       | 194.56 $\pm$ 43.91 | $-\infty$       | 424.3 $\pm$ 238.99 | $-\infty$        |
> | **BO**            | 97.26 $\pm$ 11.19      | 0.75 $\pm$ 0.03 | 99.02 $\pm$ 25.92  | 0.72 $\pm$ 0.06 | 104.80 $\pm$ 26.16 | 0.73 $\pm$ 0.05  |
> | **CCDDEA**        | 197.25 $\pm$ 3.70      | 0.00 $\pm$ 0.00 | 195.40 $\pm$ 2.62  | 0.00 $\pm$ 0.00 | 167.82 $\pm$ 44.25 | -0.02 $\pm$ 0.11 |
>
> As shown in Table 1, the experimental results demonstrate that when the data volume is small (N=100), the performance of all algorithms is significantly affected. For instance, the FS values of TT-DDEA exhibit considerable fluctuations, highlighting the difficulty of effectively optimizing the objective function in low-data scenarios. Moreover, under small data volume conditions, the SI values frequently approach 0 or $-\infty$, further confirming that insufficient data volume undermines the convergence and stability of the algorithms. We also observed that BO and ARCOO are minimally affected by data volume and are able to find better solutions, demonstrating their exceptional capabilities.
>
> > **Q3** About randomness.
>
> **A3** Thank you for your valuable feedback. In fact, the fluctuating performance of the baseline methods is due to the different random seeds used by the algorithm, rather than the task itself.

---

> > ### Author Response · Authors · 2024-11-18
> > **Reply to Reviewer Lkg5 (3/3)**
> >
> > > **Q4** About technical challenge.
> >
> > **A4** Thank you for your question. Compared to the challenge of constructing a dataset, the primary difficulty and contribution of this paper lie in identifying the **key issues unique** to offline optimization. Different from the traditional goal that is to pursue the optimal solution, this paper at first discloses that **the goal of offline optimization is to stably surpass the offline dataset during optimization procedure**. We found that arbitrarily setting a small fixed optimization step size, such as 50 steps, may lead to suboptimal solutions, whereas extending the number of steps could cause some offline optimization algorithms to degrade. This proposes an anti-degradation algorithm as a logical solution.
> >
> > Building on this understanding, we developed a benchmark that reflects these goals and indicators.
> >
> > - **Our benchmark is highly customizable**, allowing users to define parameters such as data distribution, constraints, and other experimental factors. This flexibility enables researchers to control their experiments precisely, providing a robust framework to thoroughly explore and compare algorithmic performance.
> > - We introduce a novel metric, **Stability Improvement** (SI), which has not been considered in previous work. This metric is crucial in offline optimization because we do not know the exact point to online evaluation, making stability across the entire optimization process essential.

---

> > ### Comment · Reviewer_Lkg5 · 2024-11-25
> > **Thanks for your response (cont)**
> >
> > Re: A2, yes it makes sense that data volume affects performance (although it takes like 2-3 lines of codes to adjust the volume). I was under the impression that the proposed benchmark, assuming constant data volume, was all about using which portion of data to train an offline opt model (e.g., between 10-60 percentile, between 20-70, etc.). That's why I brought up the fact that in the original design bench, the bottom 30 was used, which arguably is the hardest scenario. What I meant was that effort should instead be spent to make the offline benchmarks more challenging in a meaningful way.
> >
> > Re: A3, yes -- that means performance is super volatile. I get your point that it might not be the bench's problem, but was hoping there could be some evaluation that takes into account this high fluctuation.

---

> > > ### Author Response · Authors · 2024-11-28
> > > **Reply to Reviewer Lkg5**
> > >
> > > Thank you very much! We fully agree that data volume and high fluctuation are important for our paper, and hope to address this problem further in future work.
> > >
> > > > Q1: About considering FS together with variance.
> > >
> > > **A1**:
> > >
> > > Thank you for your valuable comments. To better illustrate this, consider the following two sequences:
> > > A: [0, 0.3, 0.4, 0.4, 0.4, 0.4, 0.3]
> > > B: [0, -0.1, 0, 0.1, -0.1, 0, 0.3]
> > >
> > > The variance of A is 0.018, while B's is 0.016. Because the FS of A and B are both 0.3 in the final optimization step, B is considered to be better due to its lower variance. However, when evaluated using SI, $SI_A$ is 0.854, significantly outperforming $SI_B$, which is only 0.028. This outcome suggests that A, with its higher variance, is indeed the preferable choice, which aligns with our intuitive judgment that it represents a better sequence than B.
> > >
> > > This discrepancy between variance and SI underscores the key distinction between stability and the stability of offline optimization. SI does not merely measure the stability of a solution in isolation, but rather its robustness in maintaining a solution that is not only stable, but also capable of identifying superior solutions.
> > >
> > > Thus, when we adopt a stricter evaluation, this definition of "stability" extends beyond traditional notions of stability. It encapsulates **the stability of the process of finding solutions that surpass the offline dataset during the optimization procedure**, rather than merely producing stable solutions (i.e., low variance), which is the true goal in offline optimization.
> > >
> > > > Q2: About p-value.
> > >
> > > **A2**:
> > >
> > > Thank you for your insightful question, and I completely understand your concern. To better clarify, let’s consider the following two sequences:
> > >
> > > A: [0, 0.1, 0.1, 0.1, 0.2, -0.2, -0.1]
> > >
> > > B: [0, -0.2, -0.1, 0.1, 0.1, 0.1, 0.2]
> > >
> > > Both A and B result in the same p-value because their values are essentially similar in distribution. However, in offline optimization, we generally consider B to be superior to A due to its relatively stable improvement throughout the optimization process. This example highlights a key limitation of the p-value as a performance evaluation tool.
> > >
> > > While the p-value accounts for both the mean and variance of the distributions, it assumes a purely statistical framework, often treating optimization results as if they were independent samples. In reality, however, the optimization process is not merely a sampling process, and the sequences we work with often exhibit **temporal dependencies—sequence changes with the number of optimization steps**—that the p-value does not capture effectively.
> > >
> > > Thus, while the p-value may provide useful information in some contexts, it fails to account for the dynamic nature of optimization and the temporal dependencies inherent in sequences. Therefore, the p-value cannot replace the SI metric.
> > >
> > > **If there are any issues, please feel free to reply, and we will respond promptly.**

---

> > > ### Author Response · Authors · 2024-12-03
> > > **Kind Reminder**
> > >
> > > Dear Reviewer Lkg5,
> > >
> > > As the deadline approaches, we sincerely hope to address your concerns and discuss the rebuttal with you further. We hope that we have explained clearly the difference between p-value, variance, and our metric. If you have any questions, please feel free to ask directly! Moreover, if you find our response satisfactory, could you please kindly consider the possibility of updating the rating? Thank you very much for your valuable suggestion.
> > >
> > > Best regards,
> > >
> > > Authors

---

> ### Author Response · Authors · 2024-11-25
> **Gentle Reminder of the Revision Deadline**
>
> Dear Reviewer Lkg5,
>
> We express our sincere gratitude for your invaluable comments and constructive feedback on our paper. We understand that you have a busy schedule, and we kindly remind you that the revision deadline is approaching. Your insights have played a pivotal role in refining our research. We have addressed your concerns in our response, including **the incorporation of new experiments (such as the different sizes of the offline datasets)** and **enhancing the clarity of SI metric**. Should you have any additional questions or unresolved issues, please do not hesitate to inform us. We are fully dedicated to promptly resolving any remaining concerns within the author-reviewer discussion period. Moreover, if you find our response satisfactory, could you please kindly consider the possibility of updating the rating. Thank you very much for your valuable suggestion.
>
> Thank you for your attention to our work, and we look forward to your response.
>
> Best regards,
>
> The Authors

---

> ### Author Response · Authors · 2024-12-01
> **Kind Reminder**
>
> Dear Reviewer Lkg5,
>
> As the deadline approaches, we sincerely hope to address your concerns and discuss the rebuttal with you further. If you have any questions, please feel free to ask directly! Moreover, if you find our response satisfactory, could you please kindly consider the possibility of updating the rating. Thank you very much for your valuable suggestion.
>
> Best regards,
>
> Authors

---

### Official Review · Reviewer_x8NF · 2024-11-09

**Soundness:** 2
**Presentation:** 3
**Contribution:** 2
**Rating:** 6
**Confidence:** 3

**Summary:**

This paper introduces SOO-Bench, a new benchmark for offline black-box (BB) model-based optimization (MBO) algorithms, which allows for evaluating the algorithm stability, i.e., the ability of an algorithm to consistently and continuously improve upon existing solutions from the offline dataset. The authors frame stability as one of the core objectives of offline BB-MBO. Offline here stands for the fact that the optimization has to rely only on historical data, without having access to evaluating new solution candidates; an important property of the offline data is usually a narrow distribution that is exhibited, possibly because of the bias induced in the data collection, such that the search space is not equally covered. This is at the same time a big challenge for the offline BB-MBO algorithms. SOO-Bench allows for customizing the narrowness of the offline data distribution and proposes a stability indicator (SI) to quantify how stable an algorithm is. The benchmark consists of various real-world tasks and datasets, including a possibility to vary the difficulty of the dataset by aforementioned customization. A range of algorithms (for unconstrained and constrained optimization) are empirically evaluated on the proposed benchmark, giving insights into their stability measures.

EDIT: I raise my score in light of the rebuttal.

**Strengths:**

The presentation of the paper is clear, with clearly stated contributions. The proposed stability indicator is a novel and useful measure for offline BB-MBO. A wide range of structurally and behaviorally different BB-MBO algorithms is evaluated, showing that they greatly vary in terms of stability.

**Weaknesses:**

The authors claim multiple times that black-box optimization requires building a surrogate model from the available data (samples and their evaluations). This is not always true, as black-box optimization is very often addressed by model-free methods, which are typically iterative randomized optimization heuristics (i.e., population-based algorithms), usually stemming from the evolutionary computation field. CMA-ES is one such example: an evolution strategy that does not learn a surrogate per se, but adapts the probability distribution by means of updating the covariance matrix. It may be helpful to clarify early in the paper that you are focusing specifically (or mostly) on model-based approaches, while explaining the inherent differences of the algorithms you are considering. Furthemore, the details of the evaluated algorithms could be made clearer; related to this, it is not clear how the algorithms are selected in the first place. It would be good to include a brief description of each evaluated algorithm, perhaps in an appendix, and explain their criteria for algorithm selection. This would help readers better understand the scope and rationale of the benchmark.

**Questions:**

Could the authors clarify what is meant by ‘a guided training agent model’ which assists classic baselines (I assume BO and CMA-ES)?
How are the algorithms that are benchmarked selected?
Finally, correcting for the OOD data is widely used in machine learning. It would be interesting to see how the value of the stability indicator of an algorithm would change after incorporating some of these techniques (e.g., for covariate shift) into the algorithm.

---

> ### Author Response · Authors · 2024-11-18
> **Reply to Reviewer x8NF**
>
> We appreciate your overall positive assessment of our contributions and are grateful for your suggestion. We have already updated the new revision based on your suggestions (we also recommend the reviewer check the global response).
>
> ***List of changes in the manuscript***:
>
> > 1. Section 1 has been revised to clarify the expression of black-box optimization (BBO).
> > 2. Appendix B.1 are revised to add the details of the evaluated algorithms and pseudocodes following Q2.
>
> > **Q1** About surrogate model.
>
> **A1** Thank you for your insightful comment. You are correct that BBO consists of model-based and model-free methods. In line 39, we mention that "most BBO methods involve constructing a surrogate model." In fact, we do not intend to overlook model-free methods, and we have recognized that the phrase "most of" may be inappropriate in this context. We have revised this in the new version of the paper to provide a more accurate expression.
>
> Our paper focuses on "offline optimization", **where surrogate models are often employed to approximate the true objective function**. As such, we concentrate more on **model-based methods**. Even model-free method like CMA-ES typically require constructing a surrogate model [1], [2], [3] in offline optimization. This is because it still relies on an approximation of the objective function to express true objective function to achieve "active" sampling.
>
> We have updated the paper to clarify the expression of BBO and ensure that our focus on model-based methods is properly highlighted.
>
> > **Q2** About the details of the evaluated algorithms.
>
> **A1** Thank you for your suggestions. We have added detailed descriptions of the evaluated algorithms and their pseudocodes in Appendix B.1, highlighted in blue.
>
> [1] Huang P, Wang H, Jin Y. Offline data-driven evolutionary optimization based on tri-training. Swarm and evolutionary computation, 2021.
>
> [2] Gong Y J, Zhong Y T, Huang H G. Offline Data-Driven Optimization at Scale: A Cooperative Coevolutionary Approach. IEEE Transactions on Evolutionary Computation, 2023.
>
> [3] Trabucco B, Geng X, Kumar A, et al. Design-bench: Benchmarks for data-driven offline model-based optimization, International Conference on Machine Learning, 2022.

---

> ### Author Response · Authors · 2024-11-25
> **Gentle Reminder of the Revision Deadline**
>
> Dear Reviewer x8NF,
>
> We sincerely appreciate the time and effort you have dedicated to reviewing our work. As the deadline for updating our manuscript is rapidly approaching, we would greatly appreciate your timely feedback on the revisions and clarifications we have provided. During the rebuttal period, we diligently addressed your concerns by providing point-to-point responses, which included **the incorporation of more details of the evaluated algorithms and pseudocodes**, and **the clarification on the expression of black-box optimization (BBO)** . Would you mind checking our response (a shortened summary, and the details) ? If you have any further questions or concerns, we would be grateful if you could let us know. Moreover, if you find our response satisfactory, could you please kindly consider the possibility of updating the rating. Thank you very much for your valuable suggestion.
>
> Thank you for your attention to our work, and we look forward to your response.
>
> Best regards,
>
> The Authors

---

> ### Author Response · Authors · 2024-12-01
> **Kind Reminder**
>
> Dear Reviewer x8NF,
>
> As the deadline approaches, we sincerely hope to address your concerns and discuss the rebuttal with you further. If you have any questions, please feel free to ask directly! Moreover, if you find our response satisfactory, could you please kindly consider the possibility of updating the rating. Thank you very much for your valuable suggestion.
>
> Best regards,
>
> Authors

---

> > ### Comment · Reviewer_x8NF · 2024-12-03
> > **Thanks for addressing the points**
> >
> > Thanks to the authors for detailed responses and addressing many concerns of mine and other reviewers. Therefore I will raise my score.

---

> > > ### Author Response · Authors · 2024-12-03
> > > **Official Comment by Authors**
> > >
> > > Thank you for your prompt and positive feedback, as well as for appreciating the value of our work and raising the score. We are delighted to have addressed your questions and incorporated all your suggested modifications into the revised version based on your comments. If you have any further questions or concerns, please don’t hesitate to reach out.

---

### Author Response · Authors · 2024-11-18
**General Response to Reviewers and Revision Submitted.**

We would like to thank the reviewers for their helpful comments. We are encouraged they find our idea to be novel (**Reviewer x8NF**, **Reviewer 9Mjy**, **Reviewer 8L7a**), the work well-motivated (**Reviewer Lkg5**), and the contribution clear and meaningful for ML (**Reviewer x8NF**, **Reviewer Lkg5**, **Reviewer 9Mjy**). We are glad they point out it is a easy-to-use framework (**Reviewer 9Mjy**, **Reviewer 8L7a**), the new indicator called Stability Improvement (SI) is novel and interest (**Reviewer x8NF**). Furthermore, there is consensus among the reviewers about the well-structured nature of our paper. In light of the reviewers' comments, we have revised our manuscript, supplementary material as detailed below.

We have carefully considered all of the feedback provided, and in response, we have made the following revisions to our manuscript to address the reviewers' concerns and suggestions:

- Enhance the description of the evaluated algorithms (Appendix B.1).
- Add different setting of our benchmark, including different task sizes and diverse distributions within the solution space (Appendix D).
- Evaluate new offline optimization algorithm CCDDEA on our benchmark (Section 6.1, Table 2, 5, 9, Appendix B.1).
- Fix minor issues on grammar and typos.

All changes have been highlighted in blue in the manuscripts.

**The definition of offline optimization:** Offline optimization aims to make full use of the offline data and finally recommend potential high-quality solutions to be applied online without actively querying the solutions online. Due to the exploration nature of global optimization, optimizers may tend to try unseen out-of distribution solutions for potentially better performance (i.e., find much better solutions than the offline dataset). As the optimization procedure proceeds, the quality of solutions may tend to decline. It is difficult to determine when to stop the optimization procedure and return the final solution without active evaluation feedback. Previous work has suggested that by specifying the number of stopping steps for optimization, good results can be achieved by adjusting the step size or modifying the model for experiments. However, this is often not feasible in real life. Therefore, we hope to maintain a steady improvement throughout the optimization process, or at least maintain stability so that the solutions at any step will not be too bad.

We want to restate our contribution:

1. **Customization:** SOO-bench allows users to extensively customize data distributions, constraints, and other parameters, providing unparalleled flexibility and control for thorough experimental analysis.
2. **Novel Metric - Stability Improvement (SI):** We introduce SI to assess algorithm stability during optimization. This metric is vital as the timing for applying optimization results online is uncertain, requiring algorithms to be stable throughout the process.
3. **Diverse Tasks:** Our benchmark includes a wide range of tasks, from simple to complex and from unconstrained to constrained, catering to various research needs.

If any of the reviewers have any further questions, we would be pleased to answer them.

---

### Meta-Review · Area_Chair_WLRz · 2024-12-19

**Metareview:**

The paper considers the problem of offline model based optimization and provides new benchmarks and stability metric to evaluate the algorithms. Offline optimization is an active and important area of research, so this work could be potential useful to the community. However, I have following concerns shared by two other reviewers.

- I found the paper's contribution to be limited. Reviewer Lkg5 pointed this out as well and authors replied one contribution as **in-depth thinking on the essential goals of offline optimization (i.e., the goal of offline optimization is to stably surpass the offline dataset during optimization procedure)**. I believe this is a basic requirement for this problem setting well known to all of us working on this problem and cannot be highlighted as a new contribution.

- One huge challenge with previous benchmarks in offline optimization (primarily design-bench and another multiobjective benchmark released recently)  has been the difficulty of installing and setting them up because of mismatched dependencies and code errors. There are multiple open issues on design-bench page describing this (https://github.com/brandontrabucco/design-bench/issues). Unfortunately,  looking at the requirements.txt and README description of the code provided with the paper doesn't seem to give much confidence about this aspect. A benchmark is most beneficial to the community if they can use this easily. This is the most important aspect authors can improve on.

**Additional Comments On Reviewer Discussion:**

Reviewer x8NF pointed out the model free methods as another class of algorithms for the problem not acknowledged by the paper. The authors response addressed the comment. Reviewer Lkg5 pointed out that the paper's contributions are lacking in new contributions. I agree with it but don't consider it to be a primary reason for rejection. Both reviewer Lkg5 and 8L7a had concerns about how the offline data distribution is selected and the stability metric itself. I agree with the concerns and didn't find the authors response convincing.

---

### Decision · Program_Chairs · 2025-01-22

Accept (Poster)